# Energy-based Epistemic Uncertainty for Graph Neural Networks

**Dominik Fuchsgruber, Tom Wollschläger, and Stephan Günnemann**
School of Computation, Information and Technology & Munich Data Science Institute
Technical University of Munich, Germany
`{d.fuchsgruber, tom.wollschlaeger, s.guennemann}@tum.de`

## Abstract

In domains with interdependent data, such as graphs, quantifying the epistemic uncertainty of a Graph Neural Network (GNN) is challenging as uncertainty can arise at different structural scales. Existing techniques neglect this issue or only distinguish between structure-aware and structure-agnostic uncertainty without combining them into a single measure. We propose GEBM, an energy-based model (EBM) that provides high-quality uncertainty estimates by aggregating energy at different structural levels that naturally arise from graph diffusion. In contrast to logit-based EBMs, we provably induce an integrable density in the data space by regularizing the energy function. We introduce an evidential interpretation of our EBM that significantly improves the predictive robustness of the GNN. Our framework is a simple and effective post hoc method applicable to any pre-trained GNN that is sensitive to various distribution shifts. It consistently achieves the best separation of in-distribution and out-of-distribution data on 6 out of 7 anomaly types while having the best average rank over shifts on *all* datasets.

## 1 Introduction

Quantifying and understanding uncertainty is crucial to developing safe and reliable machine learning systems. Many applications such as Reinforcement Learning [42], Active Learning [5, 31] or Out-of-Distribution detection [60] benefit from disentangling different facets of uncertainty [54, 17, 46]. Typically, one distinguishes between an irreducible *aleatoric* component and reducible *epistemic* factors [29]. The former is inherent to the data, while the latter is rooted in a lack of knowledge. One way to quantify epistemic uncertainty is to define a classifier-dependent density over the data domain [55]. High values indicate similarity to the training data and thus imply low epistemic uncertainty. Energy-based models (EBMs) induce this density by defining an energy function that assumes low values near the training data [38]. A common choice is the logits of the classifier as their magnitude is supposed to correlate with the model confidence [40]. However, many architectures have been shown to produce arbitrarily overconfident predictions far from the training data [28]. The negative implications for logit-based EBMs are only scarcely discussed in the literature [35, 36].

While substantial effort has been directed toward uncertainty estimation for independent and identically distributed (i.i.d.) problems [1, 24, 49, 37, 6, 22], graphs have only recently received attention within the community [71, 64, 67, 21]. There, uncertainty can arise at different structural scales, e.g., from only a single node, clusters, or global properties. Previous work only accounts for this implicitly by applying techniques from i.i.d. domains to Graph Neural Networks (GNNs) [67, 49]. A recent approach distinguishes only between structure-aware and structure-agnostic uncertainty [64] while not combining them into a single measure, which is often required in downstream applications [42, 54]. Consequently, estimates are often only sensitive to shifts that match their structural resolution. They may overfit certain anomaly types and not be reliable in general.

38th Conference on Neural Information Processing Systems (NeurIPS 2024).

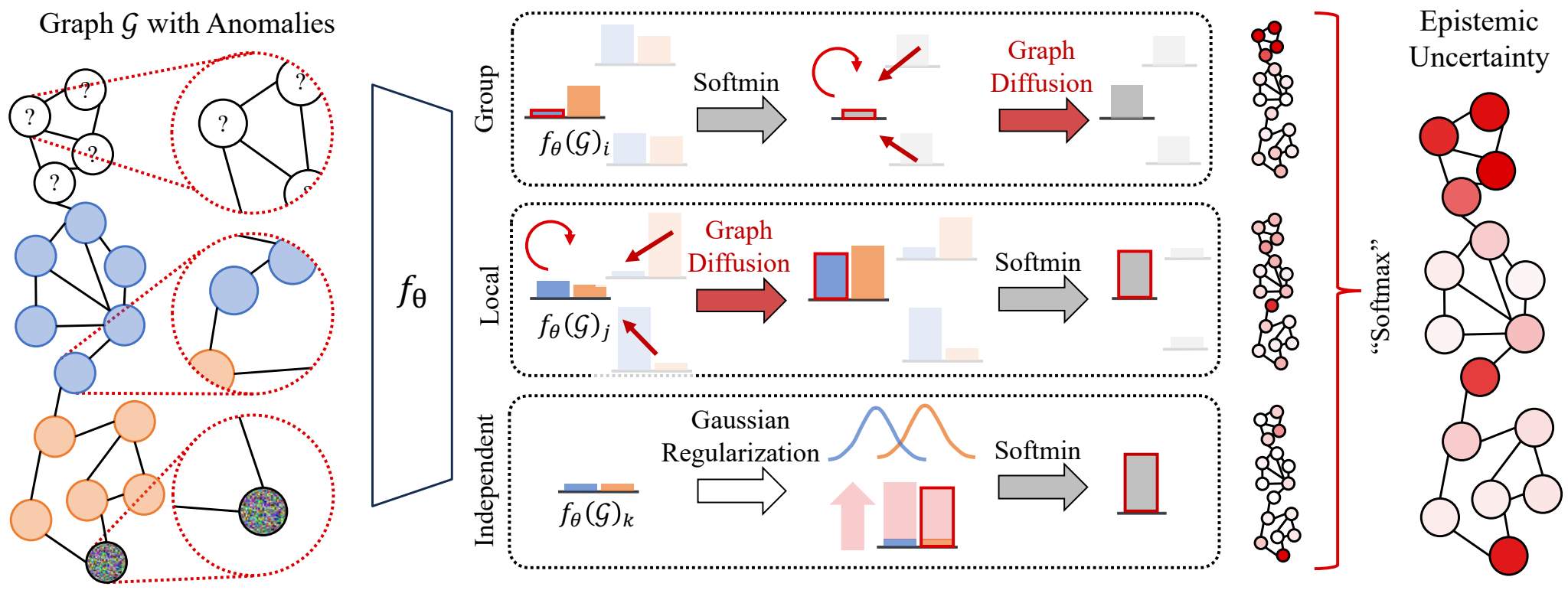

Figure 1: Overview of the Graph Energy-based Model (GEBM). Graph-agnostic energy (uncertainty) of a trained GNN $f_\theta(\mathcal{G})$ is first regularized to mitigate overconfidence and then aggregated at a local, cluster, and structure-independent scale by interleaving energy marginalization and graph diffusion. While group energy marginalizes before diffusion, local energy is fine-grained and can pick up conflicting evidence. GEBM assigns high uncertainty to several anomaly types simultaneously.

We address these shortcomings and propose a novel graph-based EBM (GEBM) that is structure-aware at different levels. The core idea behind GEBM is to define energy functions at different structural abstractions and facilitate the flexibility of the EBMs to aggregate them into a single, theoretically grounded measure. Interestingly, we find that interleaving a graph diffusion process with energy marginalization gives rise to energy functions that naturally capture patterns of different granularity. As depicted in Figure 1, we utilize three energy types: (i) *Group* energy corresponds to evidence smoothing and emphasizes anomalies clusters within the graph while not distinguishing between their type locally. (ii) *Local* energy is more granular and sensitive to evidence disagreements in the neighborhood of a node. (iii) *Independent* energy is fully structure-agnostic and describes patterns that are limited to individual nodes. Aggregating them using soft maximum selection induces a single energy measure that assigns high uncertainty to anomalies at various scales. A Gaussian regularizer provably ensures that GEBM converges to low confidence far from the training data.

We evaluate GEBM over an extensive suite of datasets, distribution shifts, and baselines[1]. It consistently exhibits state-of-the-art performance in detecting out-of-distribution (o.o.d.) instances, while other approaches are only effective in a subset of settings. On all datasets, GEBM ranks the best on average over all distribution shifts. Beyond o.o.d. detection, we discover a novel theoretical connection between EBMs and evidential models. This interpretation of GEBM enables the GNN backbone to provide accurate predictions even under severe distribution shifts.

In summary, we tackle three main deficiencies of energy-based uncertainty for graphs by:

1. Proposing GEBM, an EBM that is aware of interdependence and provides a single uncertainty estimate incorporating different structural scales.
2. Formally showing that a Gaussian regularizer mitigates the overconfidence problem of logit-based EBMs and enables GEBM to provably induce an integrable data density.
3. Showing how to interpret GEBM as an evidential approach which considerably improves the predictive robustness of the classifier under distribution shifts.

## 2 Background

**Semi-Supervised Node Classification.** We consider (semi-supervised) node classification on an attributed graph $\mathcal{G} = (\boldsymbol{A}, \boldsymbol{X})$ with $n$ nodes $\mathcal{V}$ and $m$ edges $\mathcal{E}$ that is represented by an adjacency matrix $\boldsymbol{A} = \{0,1\}^{n\times n}$ and a node feature matrix $\boldsymbol{X} \in \mathbb{R}^{n\times d}$. Each node has a label $\boldsymbol{y}_i \in \{1, \dots C\}$. Given the labels of training nodes, the task is to infer the labels of the remaining nodes. Lastly, we focus on homophilic graphs, i.e. edges are predominantly present between nodes of the same class.

[1]We provide our code at cs.cit.tum.de/daml/gebm/

**Uncertainty in Machine Learning.** Usually, uncertainty regarding the prediction of a model is disentangled into aleatoric and epistemic factors $u^{\text{alea}}$ and $u^{\text{epi}}$, respectively. The former encompasses irreducible sources of uncertainty like measurement noise and inherent ambiguities. The latter is commonly defined as the non-aleatoric components of the overall uncertainty and can in general, be reduced, for example by acquiring additional data.

**Energy-based Models.** Epistemic uncertainty is commonly defined to be anti-correlated to a classifier-dependent density $p_\theta(\boldsymbol{x})$ over the input space [55]. We study Energy-based Models (EBMs) [38] which define a joint energy $E_\theta(\boldsymbol{x}, y)$ over features and labels $p_\theta(\boldsymbol{x}, y) \propto \exp\left(-E_\theta\left(\boldsymbol{x}, y\right)\right)$. Marginalization over the labels induces a feature density with normalizer $Z_\theta$.

$$p_\theta(\boldsymbol{x}) = Z_\theta^{-1} \sum_y \exp\left(-E_\theta\left(\boldsymbol{x}, y\right)\right) = Z_\theta^{-1} \exp\left(-E_\theta(\boldsymbol{x})\right) \tag{1}$$

Intuitively, the energy $E_\theta(\boldsymbol{x}) = -\log \sum_y \exp\left(-E_\theta(\boldsymbol{x}, y)\right)$ can be interpreted as a soft minimum of the joint energies. It is low in regions of high confidence and high elsewhere. The measure of epistemic uncertainty implied by an EBM therefore is: $u^{\text{epi}}(\boldsymbol{x}) = -\log p_\theta(\boldsymbol{x}) = E_\theta(\boldsymbol{x}) + \text{const}$. We can also write the conditional class probabilities as:

$$p_\theta(y \mid \boldsymbol{x}) = \exp\left(-E_\theta\left(\boldsymbol{x}, y\right)\right) / \sum_{y'} \exp\left(-E_\theta\left(\boldsymbol{x}, y'\right)\right) \tag{2}$$

This is exactly the softmax distribution applied to $-E_\theta(\boldsymbol{x}, y)$ which directly connects the logits $f_\theta(\boldsymbol{x}) \in \mathbb{R}^C$ predicted by a classifier to the joint energy by defining $E_\theta(\boldsymbol{x}, y) := -f_\theta(\boldsymbol{x})_y$ [40].

**Evidential Deep Learning.** In contrast to first-order methods that directly predict $p(\boldsymbol{y} \mid \boldsymbol{x})$, evidential methods [62, 43] instead parameterize the corresponding second-order distribution from which first-order distributions are sampled [32]. In classification, this second-order distribution is a Dirichlet distribution with parameters $\boldsymbol{\alpha} > \mathbf{0}$ called *evidence*. They indicate the confidence of the classifier. For inference, a first-order prediction can be obtained by taking the expectation:

$$\mathbb{E}_{\boldsymbol{p} \sim \text{Dir}(\boldsymbol{\alpha})}\left[\boldsymbol{p}(\boldsymbol{y} \mid \boldsymbol{x})\right] = \boldsymbol{\alpha} / \sum_{y'} \boldsymbol{\alpha}_{y'} \tag{3}$$

## 3 Related Work

**Uncertainty Estimation for i.i.d. Data.** Disentangling uncertainty has been approached from various perspectives. A family of sampling-based approaches uses a Bayesian Information-theoretic framework [29] that relies on stochastic predictions derived from a posterior over model weights. The total uncertainty is defined on the mean predictor, epistemic uncertainty as the deviation of each sample from that mean, and aleatoric uncertainty as the difference of both estimates [1]. The posterior can either be explicitly modeled by Bayesian Neural Networks (BNNs) [6, 13, 15, 18], Monte-Carlo Dropout (MCD) [22], or implicitly realized by ensemble methods [37, 77]. Test-time augmentation [72, 73] and Stochastic Centering [66] also provide samples from this posterior. Sampling-free approaches are deterministic and estimate uncertainty with a single forward pass. Evidential methods [62, 68] predict a second-order distribution from which epistemic uncertainty is derived. Distance-based approaches quantify epistemic uncertainty as similarity to the training data [46, 48, 20, 65] and are closely related to density-based uncertainty estimation. (Deep) Gaussian Processes (GPs) [52, 45, 39] use a kernel function to measure this similarity but do not disentangle epistemic and aleatoric uncertainty. Posterior Networks combine evidential and density-based approaches by predicting density-based updates to the evidence [11, 10].

**Uncertainty Estimation for Graphs.** Many of these approaches transfer to graphs: DropEdge [59] applies MCD to edges and GPs can utilize a structure-aware kernel [53, 83, 41, 60]. SGCN [82] proposes an evidential student-teacher approach while G-$\Delta$UQ [67] applies Stochastic Centering to a GNN and improves on calibration. Graph Posterior Network (GPN) [34] diffuses the density-based evidence of a Posterior Network but provides separate measures for structure-aware and structure-agnostic uncertainty, each of which is only effective on some distribution shifts.

**Energy-based Models.** EBMs [38, 3, 2] are typically employed in generative modelling [14] but have also been applied to uncertainty estimation [16] and anomaly detection [40, 81]. To address the overconfidence of logit-based EBMs far from training data, a Gaussian regularization term has

recently been proposed [35] that we employ in our framework as well. While their work empirically validates this adjustment, we formally prove the overconfidence of logit-based EBMs to happen with high probability and Gaussian regularization to mitigate the issue. The HEAT framework [36] learns multiple corrected EBMs via stochastic gradient Langevin dynamics [76] and composes them into a single measure. In contrast, our approach does not require additional training and aggregates energy that emerges naturally at different scales in the graph from a single logit-based joint energy model. Lastly, GNNSafe [79] diffuses the logit-based energy of a GNN to improve its out-of-distribution detection. In contrast, our model considers energy beyond the cluster scale.

# 4 Method

We develop a simple yet effective **G**raph-**E**nergy-**B**ased **M**odel (GEBM) for post hoc epistemic uncertainty that is sensitive to a variety of distribution shifts from any pre-trained GNN.

## 4.1 Energy at Different Scales

Previous work either does not distinguish between uncertainty at different structural resolutions at all [79] or only disentangles structure-aware and structure-agnostic factors without combining them [64]. However, for many practical purposes, epistemic uncertainty must be quantified with a single measure [42, 5, 60]. We address these issues by proposing an EBM-based uncertainty that incorporates patterns on different natural abstractions of a graph. The density induced by GEBM, $p_\theta(\boldsymbol{x})$, serves as a singular uncertainty measure that is sensitive to multiple distribution shifts simultaneously.

Inspired by the success of graph diffusion [9, 23], we propose energy on different structural levels: (i) Graph-agnostic, for node features in isolation. (ii) Based on the evidence in the local neighborhood of a node. (iii) Within clusters in the graph. Note that defining global energy on the whole graph induces no differences at the node level and therefore we do not consider it in this work. We point to appropriate measures for a potential extension of GEBM in the existing literature [84].

We make the intriguing observation that interleaving a diffusion process $P_{\boldsymbol{A}} : \mathbb{R}^k \to \mathbb{R}^k$ with the marginalization of structure-agnostic joint energy $E_\theta(\boldsymbol{x}, y)$ as in Equation (1) induces energy functions that describe the aforementioned natural abstraction levels. Based on this insight, we can derive definitions for three types of energy functions:

**Independent Energy.** On the finest scale, we consider energy independent of structural effects by omitting the diffusion operator. This term captures uncertainty regarding node features in isolation.

$$E_{\theta,I}(\boldsymbol{x}) = -\log \sum_y \exp\left(-E_\theta(\boldsymbol{x}, y)\right) \tag{4}$$

**Local Energy.** On a coarser scale, we diffuse the joint energy *before* marginalization. This retains local information like conflicting feature-based evidence within the neighborhood of a node as the class-specific information is marginalized *after propagation*:

$$E_{\theta,L}(\boldsymbol{x}) = -\log \sum_y \exp\left[P_{\boldsymbol{A}}\left(-E_\theta(\boldsymbol{x}, y)\right)\right] \tag{5}$$

**Group Energy.** By interchanging marginalization and diffusion, we effectively smooth the marginal evidence $E_\theta(\boldsymbol{x})$. Therefore, energy gets propagated predominantly within clusters of the graph.

$$E_{\theta,G}(\boldsymbol{x}) = -P_{\boldsymbol{A}}\left[\log \sum_y \exp\left(-E_\theta(\boldsymbol{x}, y)\right)\right] \tag{6}$$

Since marginalization is done *before propagation*, less local information is preserved: The energy of a node will increase when its cluster is anomalous, regardless of whether the type of anomaly matches its own. As can be seen exemplary in Figure 1, this loss of local information comes at the benefit of being less exposed to local variability within coarser clustered patterns.

Each energy type captures anomalies that affect the corresponding structural scale. We provide synthetic experiments in Appendix C.7 as an additional intuitive explanation for the aforementioned energy terms. In practice, GEBM enables practitioners to augment our framework with further, potentially task-specific, energy functions. We empirically find however that the combination of the three naturally arising energy terms already detects a broad range of distribution shifts.

### 4.2 Regularizing Logit-based EBMs

Following Equation (1), EBMs imply a density over the data domain $p_\theta(\boldsymbol{x})$ with normalization constant:

$$Z_\theta = \int_{\mathcal{X}} \sum_y \exp\left(-E_\theta(\boldsymbol{x}, y)\right) d\boldsymbol{x} \tag{7}$$

When quantifying epistemic uncertainty with this density, the energy must be low in regions of the input feature space close to the training data, while high values should be assumed far away. To that end, $p_\theta(\boldsymbol{x})$ must be integrable, that is, have a finite normalizer $Z_\theta$. However, previous work has shown that piecewise affine classifiers which are the backbone of modern GNNs will converge to overconfident logits far from training data [28]. We remark that previous studies found GNNs to be underconfident on in-distribution data [74, 75] while the aforementioned issue of overconfidence arises from high distance to training data induced by a distribution shift (see Appendix C.8). This overconfidence is problematic, as logit-based energy can suffer from the same issue and $E_\theta(\boldsymbol{x})$ may assume arbitrarily small values far from the training data. This contradicts the aforementioned desideratum of low confidence and the implied uncertainty measure breaks under severe distribution shifts. The normalizer of the logit-based joint energy $Z_\theta$ is not finite with a high probability and therefore the EBM can not induce an integrable density $p_\theta(\boldsymbol{x})$.

**Proposition 4.1.** *Let $f_\theta : \mathbb{R}^d \to \mathbb{R}^C$ be a piecewise affine function and $\mathbb{R}^h = \bigcup_l^L Q_l$ be the disjoint set of polytopes on which $f_\theta$ is affine, i.e. $f_\theta(\boldsymbol{x}) = \boldsymbol{W}^{(l)}\boldsymbol{x} + \boldsymbol{b}^{(l)}$ for $\boldsymbol{x} \in Q_l$. Assuming the direction of the rows of each $\boldsymbol{W}^{(l)}$ to be uniformly distributed, the probability that $Z_\theta$ converges decreases exponentially in the number of non-closed linear regions $L'$ and classes $C$.*

$$\Pr\left[Z_\theta < \infty\right] = (1/2)^{C \cdot L'}$$

We provide proofs for all claims in Appendix A. Intuitively, since $f_\theta$ behaves like an affine function in the limit $\|\boldsymbol{x}\|_2 \to \infty$, its predicted logits may diverge toward $\infty$. If for any class the model produces overconfident predictions in one of its affine regions, the marginal energy $E_\theta(\boldsymbol{x})$ will diverge toward maximal confidence. As classifiers are trained to output high values for one of the classes, we expect the actual probability of a well-behaved energy function to be even lower than our theoretical bound that assumes uniform weight directions. In practice, this pathological behavior of logit-based EBMs has been observed in previous work [64, 36, 35]. We mitigate this issue by augmenting the logit-based energy and we prove that this regularization induces an integrable density.

### 4.3 Our Model

Similarly to recent work on EBMs [35], we employ a class-conditional Gaussian prior $\mathcal{N}(\boldsymbol{f}_\theta(\boldsymbol{x})_y \mid \boldsymbol{\mu}_y, \boldsymbol{\Sigma}_y)$ as a regularizer for the logit-based energy. We learn the parameters $\{\boldsymbol{\mu}_y, \boldsymbol{\Sigma}_y\}_{y=1}^C$ as the maximum likelihood estimates from the training instances of each class.

$$\hat{E}_\theta(\boldsymbol{x}, y) = E_\theta(\boldsymbol{x}, y) - \gamma \log \mathcal{N}(\boldsymbol{f}_\theta(\boldsymbol{x})_y \mid \boldsymbol{\mu}_y, \boldsymbol{\Sigma}_y) \tag{8}$$

The regularization strength $\gamma > 0$ and the choice of the diffusion operator $P_{\boldsymbol{A}}$ are the only hyperparameters of GEBM. Each of them has an intuitive interpretation: $\gamma$ controls the trade-off between the predictive dependency and distance-awareness of the EBM while $P_{\boldsymbol{A}}$ encapsulates how information is propagated over $\mathcal{G}$. Regularization ensures that the corresponding marginal density $\hat{p}_\theta(\boldsymbol{x})$ has a finite normalizer $\hat{Z}_\theta$ and is therefore integrable.

**Theorem 4.2.** *For a piecewise affine classifier $f_\theta$ as in Proposition 4.1, $\hat{p}_\theta$ is well-defined.*

$$\hat{Z}_\theta = \int_{\mathcal{X}} \sum_y \exp(-E_\theta(\boldsymbol{x}, y)) \mathcal{N}(\boldsymbol{f}_\theta(\boldsymbol{x})_y \mid \boldsymbol{\mu}_y, \boldsymbol{\Sigma}_y)^\gamma d\boldsymbol{x} < \infty$$

This regularized joint energy diverges toward maximal uncertainty in the limit. Consequently, its induced density provides an uncertainty estimator that is reliable even far away from the training data.

**Corollary 4.3.** *For a piecewise affine classifier $f_\theta$ as in Proposition 4.1, and any $\boldsymbol{x} \in \mathbb{R}^d$ almost surely:*

$$\lim_{\alpha \to \infty} \hat{p}_\theta(\alpha \boldsymbol{x}) = 0$$

We then combine the *regularized energy* $\hat{E}_\theta(\boldsymbol{x}, y)$ at different structural scales (Equations (4) to (6)).

$$\hat{E}_{\theta,\text{GEBM}}(\boldsymbol{x}) = \log\left[\exp\left(\hat{E}_{\theta,I}(\boldsymbol{x})\right) + \exp\left(\hat{E}_{\theta,L}(\boldsymbol{x})\right) + \exp\left(\hat{E}_{\theta,G}(\boldsymbol{x})\right)\right] \tag{9}$$

Since energy is defined per node, most diffusion processes (Appendix C.6) act on individual nodes as an affine function with a positive coefficient (see Appendix A.2). Therefore, each individual energy and their aggregate, $\hat{E}_{\theta,\text{GEBM}}$, induce an integrable density when using a linear diffusion operator $P_{\boldsymbol{A}}$.

**Theorem 4.4.** *For a linear diffusion operator $P_{\boldsymbol{A}}(\boldsymbol{x}) = \alpha\boldsymbol{x} + const$, $\alpha > 0$ and the regularized energy $\hat{E}_\theta(\boldsymbol{x}, y)$, GEBM induces an integrable density:*

$$\int_{\mathcal{X}} \exp\left(-\hat{E}_{\theta,GEBM}(\boldsymbol{x})\right) d\boldsymbol{x} < \infty$$

GEBM is lightweight and can be applied post hoc to *any* logit-based GNN without additional training. Its induced density is a proxy for epistemic uncertainty: $u^{\text{epi}}(\boldsymbol{x}) = -\log \hat{p}_\theta(\boldsymbol{x})$ and we compute aleatoric uncertainty as the entropy of the predictive distribution of the GNN [29]. Following [79], we realize the diffusion operator as a repeated application of a label-propagation scheme (see Appendix C.6). Intuitively, this operator is a smoothing process which is only appropriate when assuming the in-distribution data to be homophilic. In the case of non-smooth (heterophilic) training data, the proposed energy would be high for in-distribution data which is undesired behaviour for an EBM-based uncertainty estimator. To disentangle effects at different structural scales, we define each of GEBM's components on structure-agnostic regularized joint energies $\hat{E}_\theta(\boldsymbol{x}, y)$. To that end, we compute the outputs of the classifier by omitting the structure *for the computation of $\hat{E}_\theta$ only* by evaluating $f_\theta(\boldsymbol{I}, \boldsymbol{X})$, i.e. setting $\boldsymbol{A} = \boldsymbol{I}$. These outputs depend only on the node features.

### 4.4 EBMs as Evidential Models

Beyond using epistemic uncertainty to detect anomalies, we also show how the joint density induced by GEBM enables predictions that are robust against distribution shifts. First, we discover a correspondence between first-order predictions of logit-based classifiers (Equation (2)) and evidential predictions (Equation (3)). This connects the joint energy $E_\theta(\boldsymbol{x}, \boldsymbol{y})$ of an EBM to the evidence $\boldsymbol{\alpha}$:

$$\boldsymbol{\alpha} \simeq \exp\left(-E_\theta(\boldsymbol{x}, \boldsymbol{y})\right) \propto p_\theta(\boldsymbol{x}, \boldsymbol{y}) \tag{10}$$

For an integrable density $p_\theta(\boldsymbol{x}, \boldsymbol{y})$, the evidence $\boldsymbol{\alpha}$ will vanish in the limit far away from training data. Previous work on Posterior Networks [11, 10] fits a normalizing flow to obtain class-conditional densities $p_\theta(\boldsymbol{x} \mid \boldsymbol{y})$ and uses them to compute a Bayesian update to a prior evidence $\boldsymbol{\alpha}^{\text{prior}}$. Its extension to graphs, GPN [64], diffuses these structure-agnostic updates with an operator $P_{\boldsymbol{A}}$. Similarly, our GEBM framework induces a normalized joint density $\hat{p}_\theta(\boldsymbol{x}, \boldsymbol{y})$ through its regularized energy function at no additional cost. Therefore, it can also be interpreted as an evidential classifier that enables inference according to Equation (3).

$$\overbrace{\boldsymbol{\alpha}^{\text{post}} = \boldsymbol{\alpha}^{\text{prior}} + P_{\boldsymbol{A}}\left(\boldsymbol{N} * p_\theta(\boldsymbol{x}, \boldsymbol{y})\right)}^{\text{GPN [64] (Evidential)}} \qquad \overbrace{\boldsymbol{\alpha}^{\text{post}} = \boldsymbol{\alpha}^{\text{prior}} + P_{\boldsymbol{A}}\left(Z_\theta^{-1} * \exp(-\hat{E}_\theta(\boldsymbol{x}, \boldsymbol{y})\right)}^{\text{GEBM (ours)}}$$

Here, $\boldsymbol{N}$ is called an uncertainty budget and it roughly corresponds to the normalizer of the joint energy. This interpretation of EBMs recovers desirable properties of density-based evidential methods [11, 64]: Predictions will converge toward a prior $\boldsymbol{\alpha}^{\text{prior}}$ far from the training distribution. They will be less affected by anomalies in the neighborhood of a node and therefore be more robust against distribution shifts. We remark that while our framework enables this evidential inference scheme, it is also possible to instead use the backbone classifier as-is and preserve its predictive performance.

## 5 Experiments

We evaluate the efficacy of GEBM by extending the evaluation proposed in [64] and expose it to a suite of 7 distribution shifts from three families that cover a broad range of anomaly types.

### 5.1 Setup

**Datasets and Distribution Shifts.** We use seven common benchmark datasets for node classification: The five citation datasets *CoraML*[4], *CoraML-LLM*[4, 78] [4], *Citeseer* [61, 25], *PubMed* [51], *Coauthor-Physics* and *Coauthor-Computers* [63], and the co-purchase graphs *Amazon Photos* and *Amazon Computers* [44]. We expose all methods to three families of distribution shifts:

(i) **Structural**. We select nodes with the lowest *homophily* as o.o.d. and train on homophilic nodes only. This induces a shift in the local connectivity pattern of the nodes. Furthermore, we include a setting in which nodes with low *Page-Rank* (PR) centrality [56] are considered o.o.d.

(ii) **Leave-out-Class.** We withhold nodes belonging to a subset of classes during training and reintroduce them during inference. In practice, this might, for example, correspond to a new type of user joining a social network. We either select the held-out classes randomly or pick those with the lowest average homophily to make them more dissimilar to the retained classes.

(iii) **Feature Perturbations**. We randomly choose a subset of nodes and perturb their features by replacing them with noise. Since most datasets have categorical bag-of-words features, we have fine-grained control over the severity of the shift in this setting. We generate *near-o.o.d.* data by sampling from a Bernoulli distribution $\bar{p}$ fitted on the node features of the dataset $\boldsymbol{X}$. A stronger shift is induced by fixing the success probability at a value of $p = 0.5$. Drawing node features from $\mathcal{N}(0, 1)$ constitutes a domain shift (*far-o.o.d.*) for categorical data.

Existing work concerns transductive node classification which enables leakage of o.o.d. information during training [64, 79]. Instead, we study the inductive setting and remove o.o.d. nodes and their edges during training, and provide results for the transductive setting in Appendix C. All results were averaged over $5$ splits and $5$ initializations each (for standard deviations, see Appendix C).

**Model and Baselines.** We compare GEBM to different baselines for (epistemic) uncertainty estimation: Ensembles (**GCN-Ens**), MC-dropout (**GCN-MCD**), DropEdge (**GCN-DropEdge**), a combination of MC-dropout and DropEdge (**GCN-MCD+DropEdge**) and a Bayesian GCN (**BGCN**) are sampling-based approaches. We also compare against the logit-based EBM (**GCN-Energy**), **GNNSafe**, and **HEAT** as EBM baselines. Lastly, we consider **GPN** and **SGCN** as evidential methods. For all models, including GEBM, we use the same backbone architecture (see Appendix B.2).

**Metrics.** We evaluate epistemic uncertainty by detecting distribution shifts. That is, we report the AUC-ROC and AUC-PR metrics for the binary *out-of-distribution detection* problem of separating in-distribution (i.d.) from out-of-distribution (o.o.d.) nodes. Additionally, we report how well uncertainty correlates with erroneous (*misclassification detection*) in Appendix C.3. Since our proposed method does not alter the softmax predictions of the backbone model, it will affect neither the aleatoric estimates nor their *calibration*. Consequently, our framework is open to additional post hoc calibration methods such as temperature scaling [26]. For completeness, we report the Expected Calibration Error (ECE) [50] and the Brier score [7] in Appendix C.4 for the GNN backbone.

### 5.2 Results

**Out-of-Distribution Detection.** Table 1 shows the performance of different aleatoric and epistemic uncertainty methods on various distribution shifts (full results in Table 6). Across all datasets and distribution shifts, our model provides the best or second-best separation of o.o.d. data from i.d. data. In practice, an estimator that is effective on all distribution shifts simultaneously is desirable. Therefore, we rank (↓) all estimators individually for each shift and dataset and report its average rank over all distribution shifts for each dataset. To not favor one distribution shift family, we adjust the weight such that Leave-out-Class, structural shifts, and feature perturbations contribute the same amount. We rank all epistemic uncertainty proxies both against other epistemic measures and aleatoric uncertainty separately. In both cases, Tables 2 and 7 show that our proposed EBM-based epistemic uncertainty is the only estimator that is effective under different distribution shifts at the same time. On all datasets, GEBM improves the AUC-ROC scores by $5.8$ to $10.9$ percentage points on average ($16\%$-$32\%$ relative improvement) over a vanilla EBM, the second best-ranked estimator (Appendix C.2). Since all EBM-based approaches (GCN-EBM, GCNSafe, GEBM) are post hoc methods, they share the aleatoric uncertainty and high predictive accuracy of the backbone.

Out of all 7 distribution shifts, our framework is only less sensitive to the centrality shift. Many baselines only perform well because of their symmetric diffusion operator that biases any arbitrary

| | Model | LoC *(last)* AUC-ROC↑ (Alea. / Epi.) | LoC *(last)* Acc.↑ | Ber(0.5) AUC-ROC↑ (Alea. / Epi.) | Ber(0.5) Acc.↑ | $\mathcal{N}(0,1)$ *(far)* AUC-ROC↑ (Alea. / Epi.) | $\mathcal{N}(0,1)$ *(far)* Acc.↑ | Homophily AUC-ROC↑ (Alea. / Epi.) | Homophily Acc.↑ |
|---|---|---|---|---|---|---|---|---|---|
| CoraML | GCN-DE | 86.2/73.0 | 87.2 | 47.1/68.8 | 71.9 | 26.8/57.8 | 71.9 | 70.8/54.5 | 85.6 |
| | GCN-Ens | 89.8/73.9 | 90.4 | 59.4/64.1 | 75.9 | 30.4/52.7 | 75.9 | 71.5/68.5 | 91.5 |
| | GPN | 85.3/88.1 | 88.5 | 53.4/52.4 | 72.8 | 51.4/55.5 | 72.8 | 66.9/51.5 | 87.6 |
| | GCN-EBM | 89.7/89.9 | 90.3 | 59.9/58.6 | 75.8 | 31.7/25.7 | 75.8 | 71.3/71.2 | 91.2 |
| | GCN-HEAT | 89.7/87.1 | 90.3 | 59.9/64.4 | 75.8 | 31.7/76.0 | 75.8 | 71.3/70.0 | 91.2 |
| | GCNSafe | 89.7/91.6 | 90.3 | 59.9/53.4 | 75.8 | 31.7/35.3 | 75.8 | 71.3/73.1 | 91.2 |
| | **GCN-GEBM** | 89.7/**91.6** | 90.3 | 59.9/**94.5** | 75.8 | 31.7/**86.4** | 75.8 | 71.3/**76.7** | 91.2 |
| Amazon Photo | GCN-DE | 76.4/61.2 | 91.7 | 52.8/51.4 | 88.5 | 41.8/51.5 | 88.5 | 62.7/52.0 | 96.6 |
| | GCN-Ens | 74.7/80.3 | 91.9 | 52.3/51.3 | 89.9 | 42.6/55.0 | 89.9 | 62.0/65.2 | 97.5 |
| | GPN | 74.5/**87.3** | 88.8 | 55.3/54.9 | 86.5 | 54.2/55.5 | 86.5 | 65.9/47.0 | 97.1 |
| | GCN-EBM | 74.6/75.2 | 91.8 | 52.1/52.1 | 89.8 | 44.2/42.6 | 89.8 | 61.9/60.0 | 97.4 |
| | GCN-HEAT | 74.6/77.3 | 91.8 | 52.1/51.8 | 89.8 | 44.2/57.3 | 89.8 | 61.9/62.6 | 97.4 |
| | GCNSafe | 74.6/75.6 | 91.8 | 52.1/50.9 | 89.8 | 44.2/46.4 | 89.8 | 61.9/60.5 | 97.4 |
| | **GCN-GEBM** | 74.6/83.6 | 91.8 | 52.1/**64.2** | 89.8 | 44.2/**92.4** | 89.8 | 61.9/**68.3** | 97.4 |
| Coauthor-CS | GCN-DE | 89.2/73.7 | 91.3 | 64.3/67.9 | 88.8 | 33.7/60.3 | 88.8 | 62.9/55.7 | 97.1 |
| | GCN-Ens | 89.7/87.3 | 92.4 | 69.6/67.8 | 90.0 | 32.8/60.7 | 90.0 | 63.9/69.1 | 97.9 |
| | GPN | 73.7/88.3 | 86.6 | 54.1/59.7 | 81.7 | 53.6/59.9 | 81.7 | 61.3/45.5 | 97.9 |
| | GCN-EBM | 89.7/89.9 | 92.3 | 69.3/71.6 | 89.8 | 34.7/28.8 | 89.8 | 63.8/60.5 | 97.8 |
| | GCN-HEAT | 89.7/88.6 | 92.3 | 69.3/70.0 | 89.8 | 34.7/69.8 | 89.8 | 63.8/66.6 | 97.8 |
| | GCNSafe | 89.7/92.1 | 92.3 | 69.3/63.0 | 89.8 | 34.7/36.9 | 89.8 | 63.8/62.4 | 97.8 |
| | **GCN-GEBM** | 89.7/**92.8** | 92.3 | 69.3/**99.5** | 89.8 | 34.7/**90.0** | 89.8 | 63.8/**69.5** | 97.8 |

Table 1: Out-of-Distribution detection AUC-ROC (↑) using aleatoric or epistemic uncertainty (best and runner-up). Our epistemic measure achieves the strongest performance on most datasets and shifts while maintaining the classification accuracy of the GCN backbone.

| Model | CoraML | CoraML LLM | Citeseer | PubMed | Amazon Computers | Amazon Photo | Coauthor CS | Coauthor Physics |
|---|---|---|---|---|---|---|---|---|
| GCN-DE | 6.3/14.2 | 8.0/16.9 | 7.8/16.2 | 7.3/13.6 | 7.2/15.9 | 7.1/15.8 | 6.5/13.2 | 8.1/16.8 |
| GCN-Ens | 7.8/16.1 | 6.3/15.0 | 6.4/12.9 | 6.2/12.4 | 5.7/11.6 | 4.7/9.0 | 4.8/10.4 | 6.2/13.3 |
| GPN | 7.0/15.2 | 5.4/11.2 | 6.6/12.9 | 5.8/12.4 | 5.7/11.8 | 5.3/11.7 | 7.2/13.9 | 6.1/14.5 |
| GCN-EBM | 4.5/7.9 | 4.0/8.2 | 5.2/10.3 | 5.2/12.0 | 4.2/10.7 | 4.9/10.9 | 4.1/7.1 | 4.4/8.4 |
| GCN-HEAT | 4.4/10.4 | 5.4/12.9 | 3.8/9.2 | 5.6/12.8 | 3.5/8.6 | 4.5/10.2 | 4.2/8.3 | 4.7/10.4 |
| GCNSafe | 5.3/9.1 | 4.4/8.1 | 5.2/9.5 | 5.4/8.9 | 5.9/13.2 | 6.4/12.9 | 5.5/10.7 | 5.8/11.7 |
| **GCN-GEBM** | **2.7 / 4.5** | **2.9 / 4.9** | **3.3 / 5.7** | **3.8 / 7.4** | **2.5 / 4.2** | **2.7 / 4.5** | **2.7 / 4.3** | **3.0 / 5.0** |

Table 2: Average o.o.d. detection rank (↓) of epistemic uncertainty versus other epistemic measures / all uncertainty measures over all distribution shifts (best and runner-up). GEBM has the best average performance rank over all distribution shifts and epistemic (and aleatoric) uncertainty estimators.

signal toward high-degree nodes regardless of the quality of the uncertainty (see Appendix C.6). While this bias could be explicitly incorporated into GEBM by defining an additional energy term, the goal of our work is not to engineer our measure toward certain downstream settings. GEBM is already sensitive to many distribution shifts as-is. Even after accounting for this setting, our estimator achieves the best average rank over all shifts. We defer further discussion to Appendix B.

**Misclassification Detection.** In alignment with previous work [64], we observe aleatoric uncertainty to be more effective for misclassification detection than epistemic estimates. Our method performs competitively among other epistemic measures as shown in Table 16. As previously discussed, improving aleatoric uncertainty for GNNs is beyond the scope of this work.

**Robust Evidential Inference.** As described in Section 4.4, GEBM enables evidential inference. We expose our model to feature shifts sampled from $\mathcal{N}(0,1)$ and increase the fraction of perturbed nodes. Figure 2 shows the predictive performance of the classifier induced by an evidential interpretation of the EBM at different regularization weights $\gamma$ compared to a vanilla EBM and the evidential model, GPN. We maintain high accuracy like an evidential model, while the performance of the baselines rapidly deteriorates. The hyperparameter $\gamma$ can be seen as interpolating between an unregularized overconfident logit-based EBM and an uncertainty-aware evidential method. While GPN requires explicit evidential training, our method enables robust inference at negligible additional cost from a pre-trained GNN. Furthermore, GPN notably sacrifices predictive capability on clean data, while GEBM enables more control over the trade-off between accuracy on clean and perturbed data.

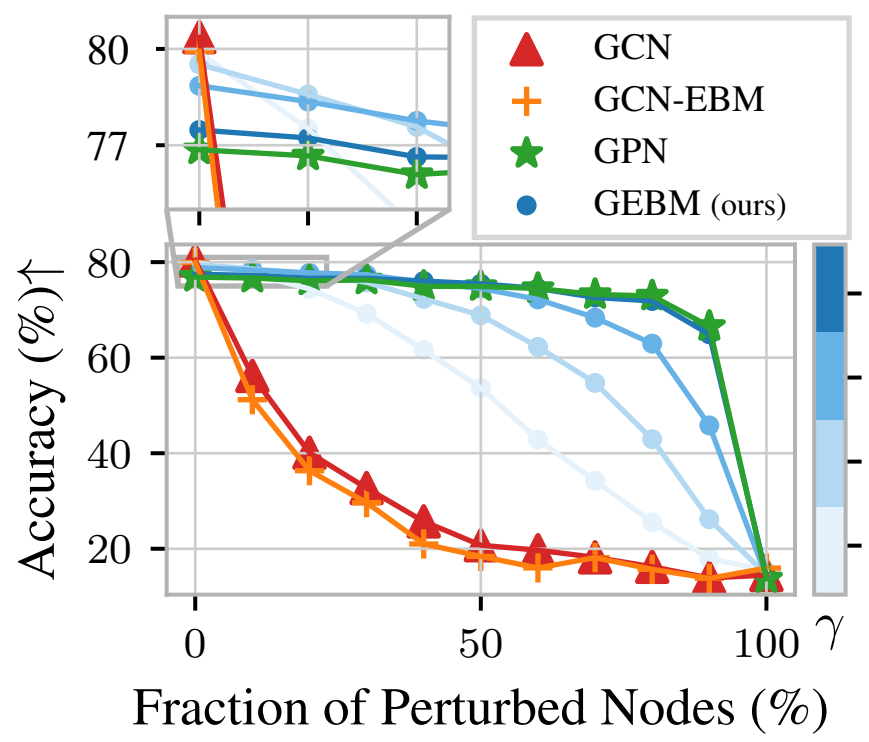

Figure 2: Accuracy of evidential inference at different regularization $\gamma$. GEBM performs on-par with evidential methods at increasingly severe perturbations.

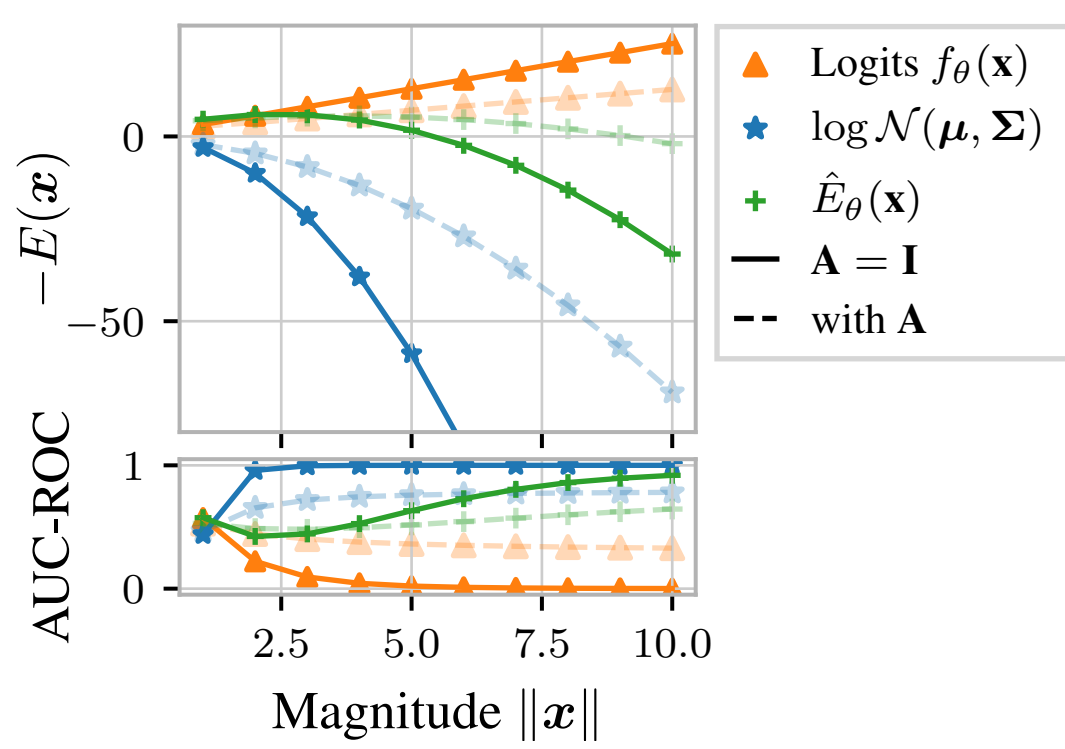

Figure 3: Logit-based, Gaussian, and regularized energy at different magnitudes. Logit-EBMs become overconfident while the corrected energy is eventually dominated by its regularizer.

The advantage of this evidential perspective is that the evidence approaches zero under increasingly severe distribution shifts. Therefore, a fixed number of diffusion steps suffices to effectively counteract the influence of anomalous neighbors when making predictions for a node. In contrast, using diffusion at, for example, the predictive level is inadequate: As shown in Proposition 4.1, the energy is likely to diverge and, therefore, the number of diffusion steps necessary to mitigate arbitrarily severe distribution shifts is unbounded. Consequentially, we find that even compared to models like APPNP that heavily rely on graph diffusion at the predictive level, the evidential interpretation of GEBM notably improves robustness (see Figure 6).

### 5.3 Ablations.

**Energy at Different Scales.** In Table 3, we compare GEBM to energy at specific structural scales and a variant without regularization ($\gamma = 0$). Each of the former corresponds to one of the energies that the GEBM framework is composed of. Group energy is equivalent to a regularized logit-based EBM which is therefore also ablated implicitly. As expected, each component is sensitive to different shifts. This confirms that interleaving marginalization and diffusion captures patterns at different resolutions. As shown in Table 26, a variant of the aggregate GEBM achieves the best rank on 7 of 8 datasets. This shows that scale-awareness is the key ingredient for effective uncertainty estimation on graphs which can be further improved with energy regularization. In far-o.o.d. settings, this regularization is crucial to obtain reliable estimates as it mitigates overconfidence issues.

| Model | LoC | $\mathcal{N}(0,1)$ | Homo. | Rank(↓) |
|---|---|---|---|---|
| EBM | 87.5 | 24.7 | 67.9 | 4.1 |
| Indep. | 79.2 | **99.7** | 63.9 | 3.7 |
| Local | 86.9 | 62.1 | **79.8** | 4.3 |
| Group | **91.7** | 62.8 | 68.9 | 3.3 |
| GEBM-$E_\theta$ | *90.1* | 1.4 | 71.1 | *3.1* |
| GEBM | 89.7 | *89.8* | *73.1* | **2.6** |

Table 3: O.o.d.-detection AUC-ROC(↑) using different EBMs (best, runner-up). GEBM combines the benefits of energy at different scales and ranks best over splits.

**Energy Regularization.** We also ablate the effect of regularizing the joint logit-based energy in Figure 3. At increasing distance from the training data, logit-based energy becomes overconfident. In contrast, the regularized $\hat{E}_\theta$ follows the logit-based energy near the training data and is dominated by the regularizer far away. This is reflected in the clear separation between perturbed and unperturbed nodes. When computing structure-aware energy, feature corruptions affect unperturbed nodes through the diffusion operation, making a clean separation difficult. This justifies our choice to compute structure-agnostic joint energy $\hat{E}_\theta(\boldsymbol{x}, \boldsymbol{y})$ and factor in the graph afterward by applying $P_{\boldsymbol{A}}$.

| | Model | LoC | Ber($\hat{p}$) | $\mathcal{N}(0,1)$ | Homo. |
|---|---|---|---|---|---|
| **GCN** | EBM | 89.9 | 67.1 | 26.4 | 71.2 |
| | Safe | **91.6** | 56.9 | 36.1 | 73.1 |
| | GEBM | **91.6** | **77.1** | **86.6** | **76.7** |
| **GAT** | EBM | 90.2 | 55.6 | 44.1 | 72.1 |
| | Safe | **91.5** | 53.2 | 45.4 | 70.3 |
| | GEBM | 85.0 | **69.3** | **76.5** | **72.6** |
| **GIN** | EBM | 76.5 | **52.3** | 42.8 | 53.2 |
| | Safe | 79.0 | 49.3 | 46.2 | 51.2 |
| | GEBM | **80.7** | 51.4 | **53.6** | **65.4** |
| **SAGE** | EBM | 74.0 | **52.7** | 42.2 | 53.2 |
| | Safe | **77.3** | 49.2 | 46.6 | 51.1 |
| | GEBM | **77.3** | 51.6 | **54.7** | **62.7** |

Table 4: O.o.d. detection AUC-ROC(↑) using different backbones. Our method is effective on all architectures.

**Backbone Architecture.** Our method can be applied post hoc to any logit-based GNN. Table 4 evaluates the o.o.d.-detection performance of our EBM framework using differ-

ent commonly used GNN backbones: GCN [34], GAT (v2) [70, 8], GIN [80] and GraphSAGE [27]. Standard deviations and AUC-PR are reported in Appendix C.5. Over different distribution shifts, our proposed approach consistently outperforms the logit-based EBM and the graph-specific GNNSafe variation. This shows the broad applicability of our framework that enables reliable high-quality epistemic uncertainty estimation from a large family of logit-based GNNs.

## 6 Limitations and Broader Impact

**Limitations.** As GEBM is a post hoc epistemic estimator, it does not improve aleatoric uncertainty or its calibration. In particular, the GCN backbone used in this work does not consistently achieve the strongest performance in both tasks. While the structural scales that arise naturally from graph diffusion cover many distribution shifts as-is, specific applications may require augmenting GEBM with additional energy terms, as we also observe for centrality-based shifts. While GEBM enables robust evidential inference, future work may build upon its paradigm of aggregating different structural scales in the graph for fully evidential methods. Lastly, we study homophilic node classification problems and leave an extension of GEBM beyond this setting to future work.

**Broader Impact.** GEBM enables cheap and simple uncertainty quantification for GNNs which we believe to contribute to the development of more reliable AI. Nonetheless, we encourage practitioners to actively reevaluate our measure of uncertainty to mitigate risks in safety-critical domains.

## 7 Conclusion

We propose GEBM, a simple and efficient EBM for post-hoc epistemic uncertainty estimation for GNNs. To the best of our knowledge, we are the first to consider uncertainty at different structural scales and address this gap by proposing a model that aggregates energy that naturally arises from graph diffusion. It consistently outperforms existing approaches over an extensive suite of datasets at detecting various distribution shifts. We formally and empirically confirm that logit-based EBMs suffer from overconfidence and prove that the regularized GEBM mitigates this issue by inducing an integrable data density. We exploit this property by discovering a link to evidential methods that enables the backbone to provide accurate predictions even under severe distribution shifts.

## Acknowledgements

We want to give special thank to Leo Schwinn and Franz Rieger for giving helpful suggestions on an early draft of the manuscript. The research presented has been performed in the frame of the RADELN project funded by TUM Georg Nemetschek Institute Artificial Intelligence for the Built World (GNI). It is further supported by the Bavarian Ministry of Economic Affairs, Regional Development and Energy with funds from the Hightech Agenda Bayern.

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

# A Proofs

## A.1 EBMs for i.i.d. Data

We first consider proofs for formal statements regarding logit-based EBMs in general (i.e. for i.i.d. data).

**Proposition 4.1.** *Let $f_\theta : \mathbb{R}^d \to \mathbb{R}^C$ be a piecewise affine function and $\mathbb{R}^h = \bigcup_l^L Q_l$ be the disjoint set of polytopes on which $f_\theta$ is affine, i.e. $f_\theta(\boldsymbol{x}) = \boldsymbol{W}^{(l)}\boldsymbol{x} + \boldsymbol{b}^{(l)}$ for $\boldsymbol{x} \in Q_l$. Assuming the direction of the rows of each $\boldsymbol{W}^{(l)}$ to be uniformly distributed, the probability that $Z_\theta$ converges decreases exponentially in the number of non-closed linear regions $L'$ and classes $C$.*

$$\Pr\left[Z_\theta < \infty\right] \approx (1/2)^{C \cdot L'}$$

*Proof.* Consider an arbitrary direction $\boldsymbol{x} \in \mathbb{R}^d$ such that $\|\boldsymbol{x}\| = 1$. As shown in [28] (Lemma 3.1), there exists some $t$ and $\alpha_0$ such that for all $\alpha > \alpha_0$ it holds that $\alpha\boldsymbol{x} \in Q_t$. We have $f_\theta(\alpha\boldsymbol{x}) = \boldsymbol{W}^{(l)}\alpha\boldsymbol{x} + \boldsymbol{b}^{(l)}$, and therefore:

$$-E_\theta(\alpha\boldsymbol{x}, y) = \boldsymbol{W}_y^{(l)}\alpha\boldsymbol{x} + \boldsymbol{b}_y^{(l)}$$

We now distinguish between two cases as $\alpha \to \infty$:

If $\boldsymbol{W}_y^{(l)}\boldsymbol{x} \geq 0$, we have:

$$\begin{aligned}
\lim_{\alpha\to\infty} \alpha\boldsymbol{W}_y^{(l)}\boldsymbol{x} &= \infty \\
\lim_{\alpha\to\infty} \alpha\boldsymbol{W}_y^{(l)}\boldsymbol{x} + \boldsymbol{b}_y^{(l)} &= \infty \\
\lim_{\alpha\to\infty} E_\theta(\alpha\boldsymbol{x}, y) &= -\infty
\end{aligned}$$

And conversely if $\boldsymbol{W}_y^{(l)}\boldsymbol{x} \leq 0$, we have:

$$\lim_{\alpha\to\infty} E_\theta(\alpha\boldsymbol{x}, y) = \infty$$

Since we assumed the direction of $\boldsymbol{W}_y^{(l)}$ to be uniformly distributed, both cases occur with probability $0.5$. Now we consider the marginal energy assigned to $\alpha\boldsymbol{x}$:

$$\begin{aligned}
E_\theta(\alpha\boldsymbol{x}) &= -\log \sum_y \exp\left(-E_\theta(\alpha\boldsymbol{x}, y)\right) \\
&\leq \alpha \min_y \left\{-E_\theta(\boldsymbol{x}, y)\right\}
\end{aligned}$$

We now analyze when the marginal energy $E_\theta(\alpha\boldsymbol{x})$ diverges toward $\infty$ and $-\infty$ respectively. If for any $y \in \{1, \dots, C\}$ we have that $\boldsymbol{W}_y^{(l)}\boldsymbol{x} \leq 0$, then:

$$\begin{aligned}
\lim_{\alpha\to\infty} E_\theta(\alpha\boldsymbol{x}) &\leq \lim_{\alpha\to\infty} \alpha \min_y \left\{E_\theta(\boldsymbol{x}, y)\right\} \\
&= -\infty
\end{aligned}$$

Since each $\boldsymbol{W}_y^{(l)}\boldsymbol{x} > 0$ with probability $1/2$, the limit diverges toward $-\infty$, with probability at most $(1/2)^C$.

Now consider some open set $Q^{(l)}$ with a non-zero measure for over which the classifier $f_\theta$ is linear. With probability at least $1/2$, for $\boldsymbol{x} \in Q^{(l)}$, $f_\theta$ diverges toward $\infty$.

We now can consider the integral over the unnormalized density implied by $f_\theta$:

$$\begin{aligned} Z_\theta &= \int_{\mathbb{R}^d} \exp(-E_\theta(\boldsymbol{x}))d\boldsymbol{x} \\ &= \sum_l \int_{Q^{(l)}} \exp(-E_\theta(\boldsymbol{x}))d\boldsymbol{x} \end{aligned}$$

For any open region $Q^{(l)}$, the energy will diverge toward $-\infty$ with probability $(1/2)^C$ and therefore:

$$\int_{Q^{(l)}} \exp(-E_\theta(\boldsymbol{x}))d\boldsymbol{x} = \infty$$

For $Z_\theta$ to be finite and well-defined, the integral over all $L'$ open linear regions with non-zero measures need to converge simultaneously, each of which happens independently with probability $(1/2)^C$. Therefore:

$$\Pr\left[Z_\theta < \infty\right] = (1/2)^{C \cdot L'}$$

□

**Remark.** Note that the assumption that the rows of each $\boldsymbol{W}^{(l)}$ have a uniformly distributed direction is given when initializing the model from a normal distribution. When training a classifier with the commonly used cross-entropy objective, the training incentivizes the model to assign high logits to exactly one of the classes for each datapoint. Therefore, the model is intuitively encouraged to have at least one row of $\boldsymbol{W}^{(l)}$ for which the $\boldsymbol{x}$ that are in the corresponding region $Q^{(l)}$ have positive projection $\boldsymbol{W}_y^{(l)}\boldsymbol{x} > 0$. In practice, we expect that for a trained model the probability of any $\boldsymbol{x}$ to have a negative projection onto all rows of the corresponding $\boldsymbol{W}^{(l)}$ to be lower than $(1/2)^C$.

**Theorem 4.2.** *For a piecewise affine classifier $f_\theta$ as in Proposition 4.1, $\hat{p}_\theta$ is well-defined.*

$$\hat{Z}_\theta = \int_{\mathcal{X}} \sum_y \exp(-E_\theta(\boldsymbol{x}, y)) \mathcal{N}(\boldsymbol{f}_\theta(\boldsymbol{x}) \mid \boldsymbol{\mu}_y, \boldsymbol{\Sigma}_y)^\gamma d\boldsymbol{x} < \infty$$

*Proof.* We consider the corrected joint energy $\hat{E}_\theta(\alpha\boldsymbol{x}, y)$ for some sufficiently large $\alpha > \alpha_0$ and an arbitrary direction $\boldsymbol{x} \in \mathbb{R}^d$.

$$\begin{aligned} \hat{E}_\theta(\alpha\boldsymbol{x}, y) &= E_\theta(\alpha\boldsymbol{x}, y) - \gamma \log \mathcal{N}(f_\theta(\alpha\boldsymbol{x})_y \mid \boldsymbol{\mu}_y, \boldsymbol{\Sigma}_y) \\ &= -f_\theta(\alpha\boldsymbol{x})_y + \frac{\gamma}{2}(f_\theta(\alpha\boldsymbol{x})_y - \boldsymbol{\mu}_y)^T \boldsymbol{\Sigma}_y^{-1} (f_\theta(\alpha\boldsymbol{x})_y - \boldsymbol{\mu}_y) + \text{const} \\ &= -\boldsymbol{W}_y^{(l)}\alpha\boldsymbol{x} - b_y^{(l)} + \frac{\gamma}{2}(\boldsymbol{W}_y^{(l)}\alpha\boldsymbol{x} + b_y^{(l)} - \boldsymbol{\mu}_y)^T \boldsymbol{\Sigma}_y^{-1} (\boldsymbol{W}_y^{(l)}\alpha\boldsymbol{x} + b_y^{(l)} - \boldsymbol{\mu}_y) + \text{const} \end{aligned}$$

For large enough $\alpha_0$, the terms quadratic in $\alpha\boldsymbol{x}$ will dominate linear terms and we can write for some $c > 0$:

$$\begin{aligned} \hat{E}_\theta(\alpha\boldsymbol{x}, y) &\geq c * (\boldsymbol{W}_y^{(l)}\alpha\boldsymbol{x})^T \boldsymbol{\Sigma}_y^{-1} (\boldsymbol{W}_y^{(l)}\alpha\boldsymbol{x}) \\ &= \alpha^2(\sqrt{c}\boldsymbol{W}_y^{(l)}\boldsymbol{x})^T \boldsymbol{\Sigma}_y^{-1} (\sqrt{c}\boldsymbol{W}_y^{(l)}\boldsymbol{x}) \\ &\geq \boldsymbol{x}^T \boldsymbol{\Sigma}_*^{-1} \boldsymbol{x} \end{aligned}$$

Here, we pick $\Sigma_*$ such that the equation holds for all $y \in \{1, \ldots, C)$. Note that this means that in the limit, each joint energy term is bounded by the logarithm of the same Gaussian $\log \mathcal{N}(\boldsymbol{0}, \boldsymbol{\Sigma}_*)$. For the marginal energy it holds:

$$
\begin{aligned}
\hat{E}_\theta(\alpha \boldsymbol{x}) &= -\log \sum_y \exp\left(-\hat{E}_\theta(\alpha \boldsymbol{x}, y)\right) \\
&\geq -\log \sum_y \exp\left(-\alpha \boldsymbol{x}^T \boldsymbol{\Sigma}_*^{-1} \alpha \boldsymbol{x}\right) \\
&= \alpha \boldsymbol{x}^T \boldsymbol{\Sigma}_*^{-1} \alpha \boldsymbol{x} - \log C \\
&\geq \alpha \boldsymbol{x}^T \boldsymbol{\Sigma}'^{-1}_* \alpha \boldsymbol{x}
\end{aligned}
$$

Here, we absorb the constant $\log C$ into the quadratic term. Similar to the proof of Proposition 4.1, we now analyze the integral of the implied density for each linear region of the classifier $Q^{(l)}$.

$$
\int_{Q^{(l)}} \exp\left(-\hat{E}_\theta(\boldsymbol{x})\right) d\boldsymbol{x} = \int_{\substack{Q^{(l)} \\ \|\boldsymbol{x}\|_2 \leq \alpha_0}} \exp\left(-\hat{E}_\theta(\boldsymbol{x})\right) d\boldsymbol{x} + \int_{\substack{Q^{(l)} \\ \|\boldsymbol{x}\|_2 > \alpha_0}} \exp\left(-\hat{E}_\theta(\boldsymbol{x})\right) d\boldsymbol{x}
$$

The first term is an integral of a finite function over a finite region, which therefore will also be finite:

$$
I^{(l)}_{\alpha \leq \alpha_0} := \int_{\substack{Q^{(l)} \\ \|\boldsymbol{x}\|_2 \leq \alpha_0}} \exp\left(-\hat{E}_\theta(\boldsymbol{x})\right) d\boldsymbol{x} < \infty
$$

For the second term, we notice that it is bounded by the (finite) Gaussian integral:

$$
\begin{aligned}
I^{(l)}_{\alpha > \alpha_0} &:= \int_{\substack{Q^{(l)} \\ \|\boldsymbol{x}\|_2 > \alpha_0}} \exp\left(-\hat{E}_\theta(\boldsymbol{x})\right) d\boldsymbol{x} \\
&\leq \int_{\substack{Q^{(l)} \\ \|\boldsymbol{x}\|_2 > \alpha_0}} \exp\left(-\boldsymbol{x}^T \boldsymbol{\Sigma}'^{-1}_* \boldsymbol{x}\right) d\boldsymbol{x} \\
&< \infty
\end{aligned}
$$

Lastly, since we can decompose $\mathbf{R}^d$ into a finite set of linear regions over each of which the integral is finite, the entire integral is finite.

$$
\begin{aligned}
\int_{\mathbb{R}^d} \exp\left(-\hat{E}_\theta(\boldsymbol{x})\right) d\boldsymbol{x} &= \sum_{l=1}^{L} \int_{Q^{(l)}} \exp\left(-\hat{E}_\theta(\boldsymbol{x})\right) d\boldsymbol{x} \\
&= \sum_{l=1}^{L} \left(I^{(l)}_{\alpha \leq \alpha_0} + I^{(l)}_{\alpha > \alpha_0}\right) \\
&< \infty
\end{aligned}
$$

□

**Theorem A.1.1.** *For a piecewise affine classifier $f_\theta$ as in Proposition 4.1, and a piecewise affine feature extractor $g_\theta(\boldsymbol{x})$, $\hat{p_\theta}$ is integrable.*

$$
\hat{Z}_\theta = \int_{\mathcal{X}} \sum_y \exp(-E_\theta(\boldsymbol{x}, y)) \, \mathcal{N}(\boldsymbol{g}_\theta(\boldsymbol{x}) \mid \boldsymbol{\mu}_y, \boldsymbol{\Sigma}_y)^\gamma d\boldsymbol{x} < \infty
$$

*Proof.* Note that both affine functions $f_\theta$ and $g_\theta$ partition the input space $\mathbb{R}^d$ into a finite set of linear regions. Intersecting their boundaries again gives rise to a finite set of linear regions. For bounded

regions, the integral over the induced density will be finite, so we discuss the unbounded regions here. For any such region $Q^{(l)}$, we have $f_\theta(\boldsymbol{x}) = \boldsymbol{w}_f^T \boldsymbol{x} + \boldsymbol{b}_f$ and $g_\theta(\boldsymbol{x}) = \boldsymbol{W}_g \boldsymbol{x} + \boldsymbol{b}_g$.

We now partition the $\mathbb{R}^d \cap Q^{(l)}$ into four regions using the kernel spaces of $\boldsymbol{w}_f$ and $\boldsymbol{W}_g$ respectively, i.e. $\mathbb{R}^d \cap Q^{(l)} = \mathbf{0}_{f+g} \cup \mathbf{0}_{f-g} \cup \mathbf{0}_{g-f} \cup \boldsymbol{I}_{f+g}$, where we define $\mathbf{0}_{f+g} := \ker(\boldsymbol{w}_f) \cap \ker(\boldsymbol{W}_g) \cap Q^{(l)}$, $\mathbf{0}_{f-g} := \ker(\boldsymbol{w}_f) \setminus \ker(\boldsymbol{W}_g) \cap Q^{(l)}$, $\mathbf{0}_{g-f} := \ker(\boldsymbol{W}_g) \setminus \ker(\boldsymbol{w}_f) \cap Q^{(l)}$ and $\boldsymbol{I}_{f+g} := \mathrm{im}(\boldsymbol{w}_f) \cap \mathrm{im}(\boldsymbol{W}_g) \cap Q^{(l)}$.

We can now decompose the integral over $Q^{(l)}$ into integrals over all four regions. Note that of these sets, only $\boldsymbol{I}_{f+g}$ has a non-zero measure (if the affine classifier is not the null function). Therefore, we only have to focus on this domain. For some $\alpha_0$ and $\boldsymbol{x} \in \boldsymbol{I}_{f+g}$, $\|\boldsymbol{x}\|_2 \geq \alpha_0$, we have:

$$\begin{aligned}
\hat{p}_\theta(\boldsymbol{x}) &\propto \exp\left(-E_\theta(\boldsymbol{x}, y)\right) \mathcal{N}(g_\theta(\boldsymbol{x}) \mid \boldsymbol{\mu}_y, \boldsymbol{\Sigma}_y) \\
&\propto \exp\left(\boldsymbol{w}_f^T \boldsymbol{x} + \boldsymbol{b}_f\right) \exp\left(-(\boldsymbol{W}_g \boldsymbol{x} + \boldsymbol{b}_g)^T \boldsymbol{\Sigma}_g^{-1} (\boldsymbol{W}_g \boldsymbol{x} + \boldsymbol{b}_g)\right) \\
&\leq \exp\left(-\boldsymbol{x}^T \boldsymbol{\Sigma}_{*,y}^{-1} \boldsymbol{x}\right)
\end{aligned}$$

For some $\boldsymbol{\Sigma}_{*,y}^{-1}$ that bounds the quadratic function in the exponential for $\|\boldsymbol{x}\|_2 \geq \alpha_0$. The integral over the region $Q^{(l)}$ now reduces to:

$$\begin{aligned}
I^{(l)} &= \int_{\boldsymbol{x} \in Q^{(l)}} \sum_y \text{const} * \exp\left(-E_\theta(\boldsymbol{x}, y)\right) \mathcal{N}(g_\theta(\boldsymbol{x}) \mid \boldsymbol{\mu}_y, \boldsymbol{\Sigma}_y) d\boldsymbol{x} \\
&= \text{const} + \int_{\substack{Q^{(l)} \\ \|\boldsymbol{x}\|_2 \geq \alpha_0}} \sum_y \text{const} * \exp\left(-E_\theta(\boldsymbol{x}, y)\right) \mathcal{N}(g_\theta(\boldsymbol{x}) \mid \boldsymbol{\mu}_y, \boldsymbol{\Sigma}_y) d\boldsymbol{x} \\
&\leq \text{const} + \int_{\substack{Q^{(l)} \\ \|\boldsymbol{x}\|_2 \geq \alpha_0}} \sum_y \text{const} * \exp\left(-\boldsymbol{x}^T \boldsymbol{\Sigma}_{*,y}^{-1} \boldsymbol{x}\right) d\boldsymbol{x} \\
&\leq \text{const} + \int_{\substack{Q^{(l)} \\ \|\boldsymbol{x}\|_2 \geq \alpha_0}} \text{const} * \exp\left(-\boldsymbol{x}^T \boldsymbol{\Sigma}_{*}^{-1} \boldsymbol{x}\right) d\boldsymbol{x} \\
&< \infty
\end{aligned}$$

Here, we pick $\boldsymbol{\Sigma}_*$ to bound the quadratic terms of all $\boldsymbol{\Sigma}_{y,*}$. As previously mentioned, the kernel spaces of $\boldsymbol{w}_f$ and $\boldsymbol{W}_g$ have zero measure and therefore do not contribute to the integral. As previously, the integral over the entire domain $\mathbb{R}^d$ decomposes into finite integrals over all $Q^{(l)}$.

$$\begin{aligned}
\int_{\mathbb{R}} \sum_y \text{const} * \exp\left(-E_\theta(\boldsymbol{x}, y)\right) \mathcal{N}(g_\theta(\boldsymbol{x}) \mid \boldsymbol{\mu}_y, \boldsymbol{\Sigma}_y) d\boldsymbol{x} &= \text{const} + \sum_{l=1}^{L} I^{(l)} \\
&< \infty
\end{aligned}$$

□

**Corollary 4.3.** *For a piecewise affine classifier $f_\theta$ as in Proposition 4.1, and any $\boldsymbol{x} \in \mathbb{R}^d$ almost surely:*

$$\lim_{\alpha \to \infty} \hat{p}_\theta(\alpha \boldsymbol{x}) = 0$$

*Proof.* As per the proof of Theorem 4.2, for sufficiently large $\alpha$ we have for some $\boldsymbol{\Sigma}_*^{-1}$:

$$\hat{E}_\theta(\alpha \boldsymbol{x}) \geq \boldsymbol{x}^T \boldsymbol{\Sigma}_*^{-1} \boldsymbol{x}$$

From there, it follows directly that:

$$
\begin{aligned}
\lim_{\alpha\to\infty} \hat{p}_\theta(\alpha \boldsymbol{x}) &= \lim_{\alpha\to\infty} \exp(-\hat{E}_\theta(\alpha \boldsymbol{x})) \\
&= \exp\left(\lim_{\alpha\to\infty} -\hat{E}_\theta(\alpha \boldsymbol{x})\right) \\
&\leq \exp\left(\lim_{\alpha\to\infty} -\boldsymbol{x}^T \boldsymbol{\Sigma}_*^{-1} \boldsymbol{x}\right) \\
&= 0
\end{aligned}
$$

□

## A.2 Graph-based Energies at Different Scales

Here, we show that local energy (Equation (5)) and group energy (Equation (6)) induce a valid probability density when using a linear diffusion operator $P_{\boldsymbol{A}}$ and regularized energy according to Equation (8). Note that independent energy (Equation (4)) is shown to induce a well-defined probability density as it is just a graph-independent regularized energy.

**Proposition A.2.1.** *For a linear diffusion operator $P_{\boldsymbol{A}}(\boldsymbol{x}) = \alpha \boldsymbol{x} + const$, $\alpha > 0$ and the regularized energy $\hat{E}_\theta(\boldsymbol{x}, y)$, the local energy $\hat{E}_L(\boldsymbol{x})$ induces a well-defined density:*

$$
\int_{\mathcal{X}} \exp(-\hat{E}_{\theta,L}(\boldsymbol{x})) d\boldsymbol{x} < \infty
$$

*Proof.*

$$
\begin{aligned}
\hat{E}_{\theta,L}(\boldsymbol{x}) &= -\log \sum_y \exp\left(P_{\boldsymbol{A}}\left(-\hat{E}_\theta(\boldsymbol{x}, y)\right)\right) \\
&= -\log \sum_y \exp\left(-\alpha \hat{E}_\theta(\boldsymbol{x}, y) + \text{const}\right) \\
&\geq -\log \sum_y \exp\left(-\alpha \boldsymbol{x}^T \boldsymbol{\Sigma}^{-1} \boldsymbol{x} + \text{const}\right) \\
&= \alpha \boldsymbol{x}^T \boldsymbol{\Sigma}^{-1} \boldsymbol{x} + \text{const} + \log C \\
&\geq \boldsymbol{x}^T \boldsymbol{\Sigma'}_*^{-1} \boldsymbol{x}
\end{aligned}
$$

Again, we absorbed the constant terms as well as $\alpha$ into the quadratic term for large enough $\|\boldsymbol{x}\|_2$ and used the quadratic bound from Theorem 4.2. From here, it is straightforward that:

$$
\int_{\mathcal{X}} \exp\left(-\hat{E}_{\theta,L}(\boldsymbol{x})\right) d\boldsymbol{x} \leq \text{const} + \int_{\mathcal{X}} \exp\left(-\boldsymbol{x}^T \boldsymbol{\Sigma'}_*^{-1} \boldsymbol{x}\right) d\boldsymbol{x} < \infty
$$

□

**Proposition A.2.2.** *For a linear diffustion operator $P_{\boldsymbol{A}}(\boldsymbol{x}) = \alpha \boldsymbol{x} + const$, $\alpha > 0$ and the regularized energy $\hat{E}_\theta(\boldsymbol{x}, y)$, the group energy $\hat{E}_G(\boldsymbol{x})$ induces a well-defined density:*

$$
\int_{\mathcal{X}} \exp(-\hat{E}_{\theta,G}(\boldsymbol{x})) d\boldsymbol{x} < \infty
$$

*Proof.*

$$
\begin{aligned}
\hat{E}_{\theta,G}(\boldsymbol{x}) &= P_{\boldsymbol{A}}\left(\hat{E}_\theta(\boldsymbol{x})\right) \\
&= \alpha \hat{E}_\theta(\boldsymbol{x}) + \text{const} \\
&\geq \alpha \boldsymbol{x}^T \boldsymbol{\Sigma}_*^{-1} \boldsymbol{x} + \text{const} \\
&\geq \boldsymbol{x}^T \boldsymbol{\Sigma'}_*^{-1} \boldsymbol{x}
\end{aligned}
$$

Again, we have used the quadratic bound of Theorem 4.2 and absorbed $\alpha > 0$ for large enough $\|\boldsymbol{x}\|_2$. From there, it follows:

$$\int_{\mathcal{X}} \exp\left(-\hat{E}_{\theta,G}(\boldsymbol{x})\right) d\boldsymbol{x} \leq \text{const} + \int_{\mathcal{X}} \exp\left(-\boldsymbol{x}^T \boldsymbol{\Sigma'}_*^{-1} \boldsymbol{x}\right) d\boldsymbol{x} < \infty$$

□

**Remark.** We want to point out that most diffusion operations, especially the ones discussed in Appendix C.6 including the label-propagation smoothing used in our experiments, are of the form $P_{\boldsymbol{A}}(\boldsymbol{x}) = \alpha\boldsymbol{x} + \text{const}$ with $\alpha > 0$. Since we only integrate over the features of a single node and keep all other features fixed, all of these diffusion processes can be expressed as a weighted sum over nodes in the graph. Since we consider unweighted graphs, there are no negative edge weights which ensures that $\alpha > 0$ (as long as some sort of self-loop is included). Therefore, linear diffusion processes can be seen as positive affine transformations.

**Theorem 4.4.** *For a linear diffusion operator $P_{\boldsymbol{A}}(\boldsymbol{x}) = \alpha\boldsymbol{x} + const$, $\alpha > 0$ and the regularized energy $\hat{E}_\theta(\boldsymbol{x}, y)$, GEBM induces a well-defined density:*

$$\int_{\mathcal{X}} \exp\left(-\hat{E}_{\theta,GEBM}(\boldsymbol{x})\right) d\boldsymbol{x} < \infty$$

*Proof.* We previously established quadratic bounds on all three constituents of the aggregate energy for large enough $\|\boldsymbol{x}\|_2$:

$$\begin{aligned}
\hat{E}_{\theta,I}(\boldsymbol{x}) &\geq \boldsymbol{x}^T \boldsymbol{\Sigma}_I^{-1} \boldsymbol{x} \\
\hat{E}_{\theta,L}(\boldsymbol{x}) &\geq \boldsymbol{x}^T \boldsymbol{\Sigma}_L^{-1} \boldsymbol{x} \\
\hat{E}_{\theta,G}(\boldsymbol{x}) &\geq \boldsymbol{x}^T \boldsymbol{\Sigma}_G^{-1} \boldsymbol{x}
\end{aligned}$$

Hence, there must be one $\boldsymbol{\Sigma}_*^{-1} \in \{\boldsymbol{\Sigma}_I^{-1}, \boldsymbol{\Sigma}_L^{-1}, \boldsymbol{\Sigma}_G^{-1}\}$ that bounds all three energy types. We now look at the GEBM model:

$$\begin{aligned}
\hat{E}_{\theta,\text{GEBM}}(\boldsymbol{x}) &= \log\left(\exp\left(\hat{E}_{\theta,I}(\boldsymbol{x})\right) + \exp\left(\hat{E}_{\theta,L}(\boldsymbol{x})\right) + \exp\left(\hat{E}_{\theta,G}(\boldsymbol{x})\right)\right) \\
&\geq \log\left(3 * \exp\left(\boldsymbol{x}^T \boldsymbol{\Sigma}_*^{-1} \boldsymbol{x}\right)\right) \\
&= \boldsymbol{x}^T \boldsymbol{\Sigma}_*^{-1} \boldsymbol{x} + \log 3 \\
&\geq \boldsymbol{x}^T \boldsymbol{\Sigma'}_*^{-1} \boldsymbol{x}
\end{aligned}$$

As before, this bounds the density induced by GEBM:

$$\int_{\mathcal{X}} \exp\left(-\hat{E}_{\theta,\text{GEBM}}(\boldsymbol{x})\right) d\boldsymbol{x} \leq \text{const} + \int_{\mathcal{X}} \exp\left(-\boldsymbol{x}^T \boldsymbol{\Sigma'}_*^{-1} \boldsymbol{x}\right) d\boldsymbol{x} < \infty$$

□

**Remark.** This directly imposes a restriction on which energies can be included in GEBM beyond the three naturally arising graph-specific terms we propose: As long as the energy term can be bounded by an energy that grows fast enough to ensure convergence, our framework accommodates it.

We also can explicitly write out the (unnormalized) GEBM density $p_{\theta,\text{GEBM}}(\boldsymbol{x})$ in terms of its constituents $p_{\theta,I}(\boldsymbol{x})$, $p_{\theta,L}(\boldsymbol{x})$ and $p_{\theta,G}(\boldsymbol{x})$.

$$p_{\theta,\text{GEBM}}(\boldsymbol{x}) \propto \frac{1}{p_{\theta,I}(\boldsymbol{x})^{-1} + p_{\theta,L}(\boldsymbol{x})^{-1} + p_{\theta,G}(\boldsymbol{x})^{-1}}$$

## B Experimental Setup

### B.1 Datasets and Distribution Shifts

| Dataset | #Nodes $n$ | #Edges $m$ | #Features $d$ | #Classes $c$ | Avg. Feature Density (%) | Homophily (%) | Edge Density $m/n^2$ (%) | Left-out-Classes | #Nodes-i.d. (Loc) $n_{\text{id}}$ |
|---|---|---|---|---|---|---|---|---|---|
| CoraML | 2995 | 16316 | 2879 | 7 | 1.75 | 78.9 | 0.18 | 3 | 1650 |
| CoraML LLM | 2995 | 16316 | 384 | 7 | n.a. | 78.9 | 0.18 | 3 | 1650 |
| Citeseer | 4230 | 10674 | 602 | 6 | 0.76 | 94.9 | 0.06 | 2 | 2977 |
| PubMed | 19717 | 88648 | 500 | 3 | n.a. | 80.2 | 0.02 | 1 | 11842 |
| Amazon Photo | 7650 | 238162 | 745 | 8 | 34.74 | 82.7 | 0.41 | 3 | 4555 |
| Amazon Computers | 13752 | 491722 | 767 | 10 | 34.84 | 77.7 | 0.26 | 4 | 10000 |
| Coauthor CS | 18333 | 163788 | 6805 | 15 | 0.88 | 80.8 | 0.05 | 5 | 9424 |
| Coauthor Physics | 34493 | 495924 | 8415 | 5 | 0.39 | 93.1 | 0.04 | 2 | 28221 |

Table 5: Statistics about the datasets used in this work.

We use eight datasets in this work that we expose to similar kinds of distribution shifts. For CoraML, we also create a version that uses continuous word embeddings instead of categorical bag-of-word features. We use the *all-MiniLM-L6-v2* sentence transformer provided by Hugging Face [78] to embed the abstracts that are provided for each paper (node) in the citation network. All datasets are taken from PyTorch Geometric [19]. They are licensed as C.C.0 1.0 (CoraML, CoraML LLM), C.C. Attribution-NonCommercial-Share Alike 3.0 (Citeseer), OdbL 1.0 (PubMed).

We expose each dataset to the same distribution shifts which can be categorized into three families:

(i) **Leave-out-Classes.** We pre-select a subset of classes and designate them as out-of-distribution. That is, we remove them from the training set and train the GNN on the remaining set of nodes. We focus on an inductive setting, where we also remove all edges linking o.o.d. left-out-class nodes to the training graph, thus preventing information leakage. In the transductive setting, o.o.d. nodes still contribute to the training signal as they may be connected to i.d. nodes. This distribution shift can be seen as occurring on a cluster level, as we introduce anomalous nodes that are likely to cluster together. We distinguish between two kinds of selection processes for the classes to be left out: We either pick the last classes in the dataset (*LoC (last)*) or choose classes with the most heterophilic connection pattern, which distinguishes them even further from i.d. classes (*LoC (hetero.)*).

(ii) **Feature Perturbations.** We select $50\%$ of nodes at random to be o.o.d. and perturb their features by replacing them with random noise. We distinguish between three perturbation types that control the similarity of node features to training data: We generate features that are similar to i.d. data (*near-o.o.d.* in the following way: For each of the $d$ features, we compute how frequent it occurs in the dataset. We then proceed to sample each of these features independently with the corresponding success probability $\hat{\boldsymbol{p}}$ from a Bernoulli $\text{Ber}(\hat{\boldsymbol{p}})$. Instead, we can also set the success probability to $p = 0.5$ for each of the features and induce a more severe distribution shift within the bag-of-words domain of the dataset. Lastly, when sampling features according to a normal distribution $\mathcal{N}(0, 1)$, we generate out-of-domain data that should, in theory, be easy to detect *far-o.o.d.*. For the CoraML-LM and the PubMed datasets, features a not bag-of-word. Therefore, the near-o.o.d shift is omitted and the far-o.o.d. shift needs to be considered within the domain of the data.

(iii) **Structural.** We induce structure-related shifts in two ways: The first option we consider is to rank all nodes according to their local homophily. That is, we compute the ratio of neighbors with the same class: $h_i = |\{v_j \in \mathcal{N}_i : \boldsymbol{y}_i = \boldsymbol{y}_j\}|/|\mathcal{N}_i|$. We then designate $50\%$ of nodes with the highest heterophily (i.e. lowest homophily) as o.o.d (*homophily*). The second structural shift ranks nodes according to their (approximate) Page Rank centrality and declares nodes with low values as o.o.d. (*Page Rank*).

In the inductive setting, we remove the o.o.d. nodes from the training graph and re-introduce them during inference. We evaluate o.o.d. detection metrics on a validation/test set that includes both i.d. and o.o.d. nodes. While the training and validation set are randomized for each split, the test set is shared across all splits to prevent data leakage. All results are reported over five different splits and five independent model weight initializations for each of them. Where appropriate, we also report standard deviations in Appendix C.

**Remarks regarding Page Rank Shifts.** We study the centrality-based shift (which heavily correlates with a degree-based shift) for completeness reasons as a similar generation process is used in [79]. However, to which extent methods should be expected to assign high uncertainty to low-degree nodes is questionable: First, centrality and/or degree are feature-irrespective quantities. In contrast to homophily (nodes of similar classes have similar features), the confidence of a classifier may not (and should not) depend on the node degree/centrality. Furthermore, many baselines use symmetric normalization (Appendix C.6) that inherently favors high-degree nodes. Therefore, an evaluation regarding node degree/centrality as a distribution shift will not test the quality of an uncertainty measure but instead favor all models that use appropriate diffusion processes for most baselines. Regardless of our concerns regarding a centrality-based shift, we report results regarding that shift as well in Appendix C. Note that while GEBM is not among the best-performing estimators in this setting, it still achieves the highest average o.o.d. detection rank overall, indicating that its merits in all other settings heavily outweigh its performance on the Page Rank shift.

### B.2 Models and Training

At the backbone of all models, we use the same GCN [34] architecture if not specified explicitly otherwise. We use one hidden layer of dimension $64$, symmetric normalization Appendix C.6, and add self-loops to the undirected (symmetric) adjacency matrix. We use ReLU nonlinearities, enable the bias term, and use dropout at $p = 0.5$.

All models are trained with the ADAM optimizer [33] with a learning rate of $10^{-3}$, weight decay of $10^{-4}$, and a cross-entropy objective. We use early stopping on the validation loss with a patience of $50$, an absolute improvement threshold of $10^{-1}$, and select the model with the best validation loss. We implement our models in PyTorch [58] and PyTorch Geometric [19] and train on two types of machines: (i) Xeon E5-2630 v4 CPU @ 2.20GHz with a NVIDA GTX 1080TI GPU and 128 GB of RAM. (ii) AMD EPYC 7543 CPU @ 2.80GHz with a NVIDA A100 GPU and 128 GB of RAM .

As the GCN backbone used in our experiments is lightweight, model training finishes within a few minutes and VRAM consumption is dominated by the datasets. Our post hoc method can be fitted and evaluated in negligible time (<1s).

For MC-Dropout [22], we use a dropout probability of $p = 0.5$ which we also use for DropEdge [59]. At inference, we evaluate $50$ samples to compute uncertainty. For ensembles, we train $10$ backbones from different weight initializations independently. The Bayesian GNN [6, 15, 18] also is evaluated with $50$ samples during inference and uses a KL-loss with weight $10^{-1}$ and is trained with learning rate $10^{-2}$ and no weight-decay. We parametrize the weight distribution using a log transform and initialize the mean and log scale to $1.0$ and $-3.0$ respectively. Weight distributions are regularized to follow a standard normal.

For GPN [64], we follow the hyperparameters suggested by the authors and use warmup training on the normalizing flow for $5$ epochs with a learning rate of $1e - 2$, no weight decay during joint training and $1e - 2$ during warmup. The flow dimension is $16$ and is composed of $10$ radial layers. The cross-entropy regularization weight is $10^{-4}$. We use Page Rank propagation for $10$ iterations at a teleport probability of $0.1$.

For SGCN [82], we use the same GCN backbone and a GDK-prior at cutoff distance $10$ and scale $\sigma = 1.0$. For teacher training, we use a learning rate of $10^{-2}$, weight decay of $5 * 10^{-4}$ and a KL-loss weight of $10^{-1}$.

For HEAT [36], we use the hyperparameters suggested by the authors. In contrast to their study on the image domain, we do not have features with spatial extent and can not use standard deviation pooling on the volume. We set the temperature parameter of the combined HEAT model to $-1$, following the suggestion of the authors. Similar to their EBM backbone, we imitate the structure of the classifier and use a 2-layer MLP with a hidden dimension of $64$ akin to the GCN backbone.

Our GEBM framework regularizes logit-based joint energy at a strength $\gamma$. For downstream tasks, this parameter can, in general, be tuned. We find that an equal weighting of predictive energy and regularization performs well: We choose $\gamma$ such that the $95\%$ quantiles of both the training logits and the representations on which the Gaussian density model is fit are in the same range. Furthermore, recent work on deterministic uncertainty argues that density-based epistemic uncertainty should not be estimated directly from the logits [46]. We follow this advice and do not fit and evaluate the

regularizer on the output layer (logits) of the GNN but instead on its penultimate layer representations. Note that for the ReLU networks we use, all formal proofs still hold in this setting as the penultimate representation of a node $v$ is also described by a piecewise affine function [28] (see A.1.1. As a smoothing operator $P_{\boldsymbol{A}}$, we employ label-propagation smoothing with $\alpha = 0.5$ for $t = 10$ iterations. Lastly, we found that aggregating the energy terms at different scales (Equation (9)) using a sum operation instead of logsumexp to perform better in practice and use it for our experiments.

In our ablation regarding different model backbones, we use the default hyperparameters from the GCN backbone (e.g. number of hidden layers). For each model additional hyperparameters are set to the following values: For GATv2 [70, 8] we use 8 heads and use summation to aggregate them. For SAGE [27], we use no normalization at the layers.

### B.3 Uncertainty Evaluation

For sampling-based approaches, we compute aleatoric and epistemic uncertainty as entropy and mutual information respectively according to [24]:

$$u^{\text{alea}} = \mathbb{E}_{\theta \sim p(\theta|\mathcal{D})} \left[ \mathbb{H} \left[ \boldsymbol{p}(y \mid \boldsymbol{x}) \right] \right]$$

$$u^{\text{epi}} = \mathbb{MI} \left[ \theta, \boldsymbol{y} \mid \boldsymbol{x}, \mathcal{D} \right] = \mathbb{H} \left[ \mathbb{E}_{\theta \sim p(\theta|\mathcal{D})} \left[ \boldsymbol{p}(y \mid \boldsymbol{x}) \right] \right] - \mathbb{E}_{\theta \sim p(\theta|\mathcal{D})} \left[ \mathbb{H} \left[ \boldsymbol{p}(y \mid \boldsymbol{x}) \right] \right]$$

For evidential approaches, we follow [64] and use the maximum softmax response $\max_c \boldsymbol{p}(y = c \mid \boldsymbol{x})$ as aleatoric uncertainty and the total evidence $\sum_c \boldsymbol{\alpha}_c$ as an epistemic estimate. Since we want to compare single measures of epistemic uncertainty, we evaluate GPN's epistemic uncertainty *in the presence of network effects*. For deterministic models (that are at the backbone of EBM-based approaches), we use the predictive entropy as a measure of aleatoric uncertainty $\mathbb{H} \left[ p(y \mid \boldsymbol{x}) \right]$ and the energy $E_\theta(\boldsymbol{x})$ as a measure of epistemic uncertainty. For all EBMs, we use a temperature of $\tau = 1.0$.

## C Additional Results

### C.1 Out-of-Distribution Detection

We report AUC-ROC and AUC-PR metrics for the o.o.d. detection problem in an inductive and transductive setting with corresponding standard deviations over all five splits and five model initializations each in Tables 6, 8, 10 and 12. Again, we report the average performance ranks are in Tables 2, 9, 11 and 13. As mentioned in Section 5.2, we consider a weighted average that does not favor any distribution shift family. That is, we assign a weight of $1/6$ to structural and leave-out-class settings and factor in feature perturbations at a weight of $1/9$ each.

In an inductive setting, GEBM outperforms all other baselines and achieves the best rank regarding both AUC-ROC and AUC-PR metrics. In the transductive setting, the two evidential methods GPN and SGCN outperform GEBM on two datasets: We argue that this is due to feature leakage as o.o.d. data is present during training and evidential models are explicitly encouraged to assign low evidence to anomalous nodes. We want to point out that it is unrealistic to assume that o.o.d. data is available in practice and therefore strongly argue in favor of the inductive scenario as a more realistic benchmark. Nonetheless, GEBM is highly effective for transductive problems as well.

### C.2 Improvement over Second Best Method

We evaluate the improvement in AUC-ROC scores of GEBM over the estimator assigned the overall second-best rank. To that end, we average the model ranks not individually for each estimator and dataset, but instead compute an average over datasets and splits simultaneously. This way, we obtain a global rank (over datasets and shifts) for each epistemic estimate. Again, to not favor certain classes of shifts, we assign weights such that each distribution shift family has an equal contribution. Based on AUC-ROC scores listed in Table 6, we list the rank of each model in Table 14:

While our approach, GEBM, also ranks the highest globally, a vanilla logit-based EBM is the next best approach. Therefore, we compute the improvement in AUC-ROC scores over this model achieved by

| Dataset | Model | LoC (last) AUC-ROC↑ (Alea. / Epi.) | Acc.↑ | LoC (hetero.) AUC-ROC↑ (Alea. / Epi.) | Acc.↑ | Ber($\hat{p}$) (near) AUC-ROC↑ (Alea. / Epi.) | Acc.↑ | Ber(0.5) AUC-ROC↑ (Alea. / Epi.) | Acc.↑ | $\mathcal{N}(0,1)$ (far) AUC-ROC↑ (Alea. / Epi.) | Acc.↑ | Page Rank AUC-ROC↑ (Alea. / Epi.) | Acc.↑ | Homophily AUC-ROC↑ (Alea. / Epi.) | Acc.↑ |
|---|---|---|---|---|---|---|---|---|---|---|---|---|---|---|---|
| CoraML | GCN-MCD | 89.7±0.7/49.9±2.3 | 90.4±2.0 | 87.8±1.8/50.3±2.6 | 87.4±1.3 | 66.4±1.6/49.2±2.0 | 75.6±2.3 | 58.9±4.5/50.4±2.6 | 75.6±2.3 | 31.4±1.7/48.7±2.4 | 75.6±2.3 | 72.9±1.1/50.2±2.7 | 82.3±1.5 | 71.4±0.8/50.2±2.9 | 91.4±1.8 |
| | GCN-DE | 86.2±1.1/73.0±1.9 | 87.2±1.7 | 83.1±2.2/66.8±2.1 | 82.0±2.3 | 64.3±2.4/54.0±2.7 | 71.9±2.7 | 47.1±4.9/68.8±3.1 | 71.9±2.7 | 26.8±1.9/57.8±2.6 | 71.9±2.7 | 66.9±1.6/69.7±1.4 | 79.3±2.1 | 70.8±1.4/54.5±1.4 | 85.6±1.9 |
| | GCN-MCD+DE | 86.2±1.1/72.6±1.9 | 87.3±1.6 | 83.9±2.1/66.6±2.2 | 83.0±2.2 | 64.7±2.5/53.7±2.4 | 72.1±3.1 | 47.0±4.1/68.8±2.7 | 72.1±3.1 | 27.8±2.0/59.0±2.3 | 72.1±3.1 | 66.9±1.6/69.7±1.5 | 78.9±2.2 | 70.9±0.9/54.4±1.2 | 85.9±2.1 |
| | GCN-BNN | 87.5±1.4/86.0±1.4 | 89.0±1.4 | 83.8±3.0/82.7±2.7 | 84.9±3.0 | 64.6±1.8/60.9±2.9 | 73.1±2.3 | 49.9±4.8/*70.7*±3.5 | 73.1±2.3 | 32.6±1.9/56.4±2.4 | 73.1±2.3 | 69.1±1.4/59.1±2.5 | 78.9±1.8 | 71.8±0.8/75.0±1.1 | 88.6±1.9 |
| | GCN-Ens | 89.8±0.6/73.9±4.6 | 90.4±2.0 | 87.7±2.0/66.4±5.8 | 87.3±1.0 | 66.6±1.5/49.5±1.5 | 75.9±2.3 | 59.4±3.6/64.1±3.0 | 75.9±2.3 | 30.4±1.2/52.7±1.8 | 75.9±2.3 | 73.0±1.1/59.2±3.7 | 82.7±1.3 | 71.5±0.7/68.5±2.5 | 91.5±1.8 |
| | GPN | 85.3±2.0/88.1±1.4 | 88.5±1.4 | 85.1±4.4/82.9±3.1 | 86.4±2.7 | 54.1±1.6/53.4±2.5 | 72.8±3.0 | 53.4±1.9/52.4±2.5 | 72.8±3.0 | 51.4±1.3/55.5±3.5 | 72.8±3.0 | 52.4±1.8/67.1±2.6 | 78.4±3.2 | 66.9±4.0/51.5±3.7 | 87.6±2.0 |
| | SGCN | 89.2±1.1/88.1±2.2 | 90.2±1.9 | *89.8*±1.9/88.9±4.2 | 89.9±1.4 | 54.9±5.9/52.9±6.4 | 36.2±24.1 | 39.1±13.0/69.7±1.7 | 36.2±24.1 | 29.1±2.0/45.0±10.9 | 36.2±24.1 | *74.1*±5.2/57.9±20.9 | 74.8±22.3 | 72.0±0.8/*75.1*±1.9 | 90.0±1.4 |
| | GCN-EBM | 89.7±0.6/*89.9*±0.9 | 90.3±1.9 | 87.6±2.0/88.4±2.3 | 87.4±1.3 | 67.2±1.7/*67.3*±1.9 | 75.8±2.2 | 59.9±3.3/58.6±4.6 | 75.8±2.2 | 31.7±1.5/25.7±1.3 | 75.8±2.2 | 73.0±1.1/*74.2*±2.1 | 82.6±1.5 | 71.3±0.9/71.2±1.6 | 91.2±1.6 |
| | GCN-HEAT | 89.7±0.6/87.1±2.3 | 90.3±1.9 | 87.6±2.0/86.2±3.5 | 87.4±1.3 | 67.2±1.7/65.5±1.4 | 75.8±2.2 | 59.9±3.3/64.4±3.8 | 75.8±2.2 | 31.7±1.5/*76.0*±2.2 | 75.8±2.2 | 73.0±1.1/70.4±2.4 | 82.6±1.5 | 71.3±0.9/70.0±1.5 | 91.2±1.6 |
| | GCNSafe | 89.7±0.6/**91.6**±0.7 | 90.3±1.9 | 87.6±2.0/**90.5**±1.8 | 87.4±1.3 | 67.2±1.7/57.0±1.3 | 75.8±2.2 | 59.9±3.3/53.4±3.1 | 75.8±2.2 | 31.7±1.5/35.3±1.4 | 75.8±2.2 | 73.0±1.1/50.6±0.6 | 82.6±1.5 | 71.3±0.9/73.1±1.5 | 91.2±1.6 |
| | **GCN-GEBM** | 89.7±0.6/**91.6**±0.8 | 90.3±1.9 | 87.6±2.0/89.5±1.7 | 87.4±1.3 | 67.2±1.7/**77.7**±2.0 | 75.8±2.2 | 59.9±3.3/**94.5**±2.4 | 75.8±2.2 | 31.7±1.5/**86.4**±3.6 | 75.8±2.2 | 73.0±1.1/57.7±1.3 | 82.6±1.5 | 71.3±0.9/**76.7**±1.3 | 91.2±1.6 |
| CoraML-LLM | GCN-MCD | 87.1±1.0/49.9±2.2 | 90.1±1.0 | 86.6±3.5/49.1±2.3 | 89.5±2.3 | n.a./n.a. | n.a. | 70.7±1.2/50.0±2.3 | 76.9±1.4 | 37.8±1.9/49.7±2.1 | 76.9±1.4 | *70.5*±1.4/50.1±2.3 | 84.0±1.5 | 70.2±0.8/50.2±2.0 | 91.0±0.7 |
| | GCN-DE | 84.8±1.3/70.9±2.8 | 87.8±1.2 | 85.2±2.4/70.2±3.1 | 87.9±2.2 | n.a./n.a. | n.a. | 74.7±1.7/32.7±2.6 | 75.6±1.5 | 35.9±1.6/63.5±1.8 | 75.6±1.5 | 64.4±1.6/69.2±1.6 | 81.8±2.2 | 68.7±1.1/58.1±2.2 | 88.5±1.1 |
| | GCN-MCD+DE | 84.5±1.3/71.3±2.3 | 88.1±1.4 | 85.2±3.7/70.7±3.1 | 87.9±1.9 | n.a./n.a. | n.a. | *75.1*±1.5/32.7±3.2 | 75.2±2.1 | 35.4±1.7/63.7±1.8 | 75.2±2.1 | 64.7±1.0/70.1±1.1 | 81.3±1.8 | 68.7±0.9/58.9±1.4 | 88.7±1.0 |
| | GCN-BNN | 86.6±1.7/80.2±2.8 | 90.0±1.2 | 86.2±2.9/81.8±2.0 | 89.3±2.2 | n.a./n.a. | n.a. | 71.5±1.6/37.3±5.2 | 76.8±1.5 | 39.9±1.9/67.6±2.2 | 76.8±1.5 | 68.4±1.3/56.0±2.0 | 83.7±1.6 | 71.6±1.1/*72.8*±1.4 | 90.8±0.9 |
| | GCN-Ens | 87.1±0.9/79.5±4.2 | 90.1±1.0 | 86.7±3.3/76.8±5.4 | 89.5±2.4 | n.a./n.a. | n.a. | 70.8±1.6/46.5±5.6 | 77.3±1.3 | 35.8±2.1/*68.4*±1.8 | 77.3±1.3 | 70.4±1.4/63.2±1.7 | 84.2±1.2 | 70.2±0.7/69.2±1.5 | 91.3±0.6 |
| | GPN | 83.6±2.5/88.2±1.5 | 89.4±0.9 | 86.2±2.4/86.7±2.0 | 88.1±2.4 | n.a./n.a. | n.a. | 56.4±2.5/57.2±2.7 | 75.8±2.3 | 53.7±2.8/58.6±2.5 | 75.8±2.3 | 52.6±1.3/66.3±2.2 | 81.3±1.9 | 68.0±4.4/53.4±3.3 | 90.8±1.2 |
| | SGCN | 87.4±1.1/86.0±1.4 | 88.9±0.8 | 86.2±2.1/85.0±3.5 | 89.8±1.9 | n.a./n.a. | n.a. | 66.3±6.2/56.3±9.0 | 68.1±18.7 | 36.8±3.5/40.4±6.0 | 68.1±18.7 | 69.5±1.4/65.4±12.5 | 78.9±14.6 | 64.9±7.6/66.1±4.5 | 80.2±20.4 |
| | GCN-EBM | 86.9±0.9/87.1±0.9 | 90.2±0.9 | 86.7±3.3/87.5±3.3 | 89.4±2.3 | n.a./n.a. | n.a. | 70.8±1.4/70.0±1.6 | 77.1±1.3 | 35.8±2.0/33.0±2.5 | 77.1±1.3 | 70.2±1.4/**72.7**±1.4 | 83.7±1.5 | 70.1±0.7/69.7±0.7 | 91.1±0.7 |
| | GCN-HEAT | 86.9±0.9/85.8±2.4 | 90.2±0.9 | 86.7±3.3/84.6±4.0 | 89.4±2.3 | n.a./n.a. | n.a. | 70.8±1.4/71.0±3.8 | 77.1±1.3 | 35.8±2.0/65.9±6.9 | 77.1±1.3 | 70.2±1.4/62.1±3.5 | 83.7±1.5 | 70.1±0.7/68.2±2.4 | 91.1±0.7 |
| | GCNSafe | 86.9±0.9/**89.8**±1.0 | 90.2±0.9 | 86.7±3.3/**91.0**±1.4 | 89.4±2.3 | n.a./n.a. | n.a. | 70.8±1.4/59.6±1.6 | 77.1±1.3 | 35.8±2.0/42.0±3.2 | 77.1±1.3 | 70.2±1.4/50.7±0.5 | 83.7±1.5 | 70.1±0.7/71.3±1.0 | 91.1±0.7 |
| | **GCN-GEBM** | 86.9±0.9/*88.4*±1.1 | 90.2±0.9 | 86.7±3.3/*89.7*±2.0 | 89.4±2.3 | n.a./n.a. | n.a. | 70.8±1.4/**82.2**±3.8 | 77.1±1.3 | 35.8±2.0/**85.3**±3.4 | 77.1±1.3 | 70.2±1.4/51.9±0.7 | 83.7±1.5 | 70.1±0.7/**75.0**±1.2 | 91.1±0.7 |
| Citeseer | GCN-MCD | 86.7±2.4/50.7±2.4 | 84.8±1.2 | 88.3±1.5/50.1±2.5 | 90.9±0.6 | *66.0*±1.1/49.1±2.0 | 76.4±1.5 | 53.0±3.9/49.5±1.9 | 76.4±1.5 | 29.4±1.6/50.3±2.1 | 76.4±1.5 | 65.6±1.9/49.2±2.2 | 78.4±1.1 | 51.9±2.0/50.2±1.9 | 77.8±1.9 |
| | GCN-DE | 81.0±2.7/69.8±2.4 | 81.0±1.2 | 83.7±1.5/69.1±2.8 | 86.4±1.0 | 63.2±1.8/56.6±2.2 | 72.9±1.6 | 49.1±4.5/79.3±1.6 | 72.9±1.6 | 24.8±1.2/62.0±1.8 | 72.9±1.6 | 59.3±1.7/59.2±1.8 | 75.6±1.4 | 51.8±1.7/49.8±1.5 | 74.3±1.8 |
| | GCN-MCD+DE | 81.8±3.2/70.8±2.8 | 81.3±1.2 | 84.0±1.7/69.5±2.0 | 87.1±1.1 | 62.7±1.7/57.5±1.8 | 73.0±1.5 | 50.3±5.0/79.6±2.1 | 73.0±1.5 | 24.4±1.3/62.4±1.8 | 73.0±1.5 | 59.3±1.7/59.9±1.3 | 75.0±1.8 | 51.5±1.8/50.2±1.4 | 74.6±1.6 |
| | GCN-BNN | 86.3±1.6/86.6±1.5 | 84.5±1.3 | 87.3±1.8/86.6±1.8 | 90.3±1.0 | 64.7±1.2/54.3±5.3 | 75.8±1.5 | 43.7±8.5/72.7±8.8 | 75.8±1.5 | 27.6±1.9/61.2±2.0 | 75.8±1.5 | 63.2±1.9/55.9±1.9 | 77.8±0.6 | 52.8±1.9/54.1±2.0 | 77.1±2.0 |
| | GCN-Ens | 87.0±2.2/75.5±4.5 | 84.8±1.0 | 88.4±1.6/77.7±3.1 | 91.1±0.5 | 65.6±1.0/54.8±2.1 | 76.9±1.6 | 54.9±2.4/*82.4*±1.8 | 76.9±1.6 | 29.1±1.3/57.7±1.8 | 76.9±1.6 | 65.6±1.8/48.0±1.9 | 78.6±1.0 | 51.9±2.0/**55.8**±1.4 | 77.7±1.8 |
| | GPN | 88.0±2.3/88.4±2.2 | 85.6±1.6 | **90.1**±1.6/84.3±3.1 | 91.6±1.2 | 58.1±1.7/54.4±1.5 | 76.7±1.9 | 53.4±4.7/56.5±6.0 | 76.7±1.9 | 50.9±2.0/61.9±2.0 | 76.7±1.9 | 46.7±1.7/53.1±2.1 | 78.3±2.3 | 53.9±2.0/50.3±1.9 | 77.4±2.2 |
| | SGCN | 86.1±1.9/84.9±3.4 | 85.0±1.1 | 88.0±1.5/85.4±1.9 | 90.4±0.7 | 64.5±1.2/61.8±1.6 | 75.8±1.4 | 63.6±3.6/64.1±5.6 | 75.8±1.4 | 28.1±1.2/25.6±1.7 | 75.8±1.4 | 63.7±1.7/60.8±2.5 | 79.3±1.1 | 52.1±1.9/52.9±2.1 | 76.6±2.0 |
| | GCN-EBM | 86.8±2.3/87.4±2.5 | 84.8±1.0 | 88.3±1.6/88.0±1.9 | 90.9±0.7 | **66.2**±1.5/64.7±1.5 | 76.6±1.5 | 55.1±6.1/43.8±6.5 | 76.6±1.5 | 29.6±2.0/23.5±1.6 | 76.6±1.5 | *65.7*±1.8/**66.4**±2.0 | 78.7±1.1 | 51.9±2.0/52.0±2.1 | 77.5±1.9 |
| | GCN-HEAT | 86.8±2.3/85.3±2.6 | 84.8±1.0 | 88.3±1.6/86.5±2.4 | 90.9±0.7 | **66.2**±1.5/64.2±1.6 | 76.6±1.5 | 55.1±6.1/80.6±2.3 | 76.6±1.5 | 29.6±2.0/**81.2**±1.8 | 76.6±1.5 | *65.7*±1.8/61.3±2.2 | 78.7±1.1 | 51.9±2.0/53.1±1.7 | 77.5±1.9 |
| | GCNSafe | 86.8±2.3/**90.0**±2.2 | 84.8±1.0 | 88.3±1.6/*89.3*±1.8 | 90.9±0.7 | **66.2**±1.5/64.7±1.5 | 76.6±1.5 | 55.1±6.1/46.0±4.3 | 76.6±1.5 | 29.6±2.0/32.3±2.0 | 76.6±1.5 | *65.7*±1.8/43.3±1.8 | 78.7±1.1 | 51.9±2.0/*54.3*±1.9 | 77.5±1.9 |
| | **GCN-GEBM** | 86.8±2.3/*88.9*±2.1 | 84.8±1.0 | 88.3±1.6/88.7±1.4 | 90.9±0.7 | **66.2**±1.5/*66.0*±1.2 | 76.6±1.5 | 55.1±6.1/**96.3**±2.0 | 76.6±1.5 | 29.6±2.0/*72.1*±3.9 | 76.6±1.5 | *65.7*±1.8/48.3±1.8 | 78.7±1.1 | 51.9±2.0/53.7±2.0 | 77.5±1.9 |
| PubMed | GCN-MCD | 66.0±1.4/50.1±0.7 | 92.3±1.8 | 68.0±2.2/49.7±1.1 | 82.2±0.8 | n.a./n.a. | n.a. | 58.2±5.7/49.9±0.9 | 76.9±1.5 | 29.7±0.9/49.7±0.7 | 76.9±1.5 | 67.4±2.2/50.0±1.1 | 78.7±0.9 | 54.3±1.3/49.8±1.0 | 83.3±1.9 |
| | GCN-DE | 64.4±2.3/57.3±1.8 | 91.8±1.9 | 64.0±3.3/57.8±3.6 | 79.2±2.0 | n.a./n.a. | n.a. | 55.1±5.3/65.5±2.5 | 74.9±1.7 | 25.9±1.1/48.4±1.5 | 74.9±1.7 | 59.3±1.9/70.0±1.3 | 76.8±1.6 | 54.6±1.1/46.7±0.8 | 80.9±2.3 |
| | GCN-MCD+DE | 64.0±2.2/57.2±2.1 | 91.7±1.9 | 64.2±4.4/58.0±3.6 | 79.3±1.6 | n.a./n.a. | n.a. | 55.1±8.8/65.4±4.4 | 74.9±1.6 | 26.1±1.4/49.3±2.1 | 74.9±1.6 | 59.3±1.9/70.0±1.1 | 77.1±1.3 | 54.7±1.2/47.0±1.1 | 80.7±2.4 |
| | GCN-BNN | 64.5±3.0/63.9±3.9 | 92.2±1.7 | 67.4±2.2/66.5±3.7 | 81.7±3.1 | n.a./n.a. | n.a. | 53.9±6.7/67.6±4.4 | 76.4±1.1 | 29.5±1.3/56.1±2.9 | 76.4±1.1 | 64.6±3.3/51.9±4.5 | 77.1±3.3 | 54.6±1.3/57.6±1.9 | 82.8±2.2 |
| | GCN-Ens | 66.1±1.4/51.0±5.8 | 92.3±1.8 | 68.0±2.3/50.4±7.5 | 82.2±0.8 | n.a./n.a. | n.a. | 58.2±5.6/*74.7*±3.2 | 76.9±1.5 | 28.1±0.8/57.5±1.4 | 76.9±1.5 | 67.5±2.2/50.2±2.5 | 78.7±0.8 | 54.3±1.4/*59.2*±2.4 | 83.3±1.8 |
| | GPN | 62.5±4.1/62.6±2.6 | 92.9±0.8 | 67.3±4.4/67.3±3.8 | 85.0±1.3 | n.a./n.a. | n.a. | 58.4±3.8/54.7±3.0 | 76.9±1.1 | 56.0±1.0/69.2±3.0 | 76.9±1.1 | 47.5±1.0/71.1±2.2 | 80.2±1.1 | **59.3**±2.4/45.7±2.1 | 86.2±1.5 |
| | SGCN | 64.6±1.8/65.3±2.1 | 92.7±1.1 | 67.0±2.0/67.3±3.3 | 83.8±1.2 | n.a./n.a. | n.a. | 61.7±4.4/56.0±6.3 | 77.2±1.4 | 30.1±1.0/28.3±1.2 | 77.2±1.4 | 68.3±1.7/**75.7**±2.2 | 80.8±1.0 | 54.3±1.2/52.1±2.0 | 83.7±1.9 |
| | GCN-EBM | 66.0±1.5/65.8±1.7 | 92.3±1.8 | *68.2*±2.2/67.8±2.4 | 82.2±0.8 | n.a./n.a. | n.a. | 58.0±5.4/48.6±4.5 | 76.9±1.4 | 29.0±0.9/27.5±1.1 | 76.9±1.4 | 67.5±2.2/*72.6*±2.3 | 78.6±0.9 | 54.3±1.3/52.2±1.5 | 83.3±2.0 |
| | GCN-HEAT | 66.0±1.5/64.3±2.1 | 92.3±1.8 | *68.2*±2.2/63.5±5.2 | 82.2±0.8 | n.a./n.a. | n.a. | 58.0±5.5/53.2±7.3 | 76.9±1.4 | 29.0±0.9/*73.1*±13.2 | 76.9±1.4 | 67.5±2.2/58.4±6.2 | 78.6±0.9 | 54.3±1.3/56.5±2.7 | 83.3±2.0 |
| | GCNSafe | 66.0±1.5/*66.7*±2.1 | 92.3±1.8 | *68.2*±2.2/**71.8**±2.6 | 82.2±0.8 | n.a./n.a. | n.a. | 58.0±5.5/49.8±2.2 | 76.9±1.4 | 29.0±0.9/39.1±0.6 | 76.9±1.4 | 67.5±2.2/49.7±1.2 | 78.6±0.9 | 54.3±1.3/58.7±1.1 | 83.3±2.0 |
| | **GCN-GEBM** | 66.0±1.5/**70.9**±1.0 | 92.3±1.8 | *68.2*±2.2/67.2±5.1 | 82.2±0.8 | n.a./n.a. | n.a. | 58.0±5.5/**84.0**±7.4 | 76.9±1.4 | 29.0±0.9/**75.7**±4.2 | 76.9±1.4 | 67.5±2.2/49.4±1.6 | 78.6±0.9 | 54.3±1.3/59.0±1.7 | 83.3±2.0 |
| Amazon Computers | GCN-MCD | 73.9±6.0/50.0±1.1 | 88.8±1.8 | 87.0±1.4/49.7±1.1 | 94.2±1.1 | 56.8±0.6/50.1±1.0 | 80.2±2.5 | 55.2±1.2/50.1±0.9 | 80.2±2.5 | 41.7±0.7/50.1±0.9 | 80.2±2.5 | 80.8±1.0/50.1±1.1 | 81.0±1.5 | 61.9±0.8/50.1±0.7 | 91.2±1.3 |
| | GCN-DE | 75.2±5.7/73.6±3.0 | 87.9±1.9 | *87.2*±3.3/78.8±1.9 | 93.4±1.3 | 57.2±0.6/51.7±1.4 | 78.8±2.5 | 55.8±2.2/51.3±1.2 | 78.8±2.5 | 41.1±1.3/53.0±1.5 | 78.8±2.5 | 78.9±1.1/78.7±0.7 | 80.4±1.3 | 63.2±0.7/53.5±0.5 | 89.5±1.4 |
| | GCN-MCD+DE | 75.0±4.4/73.2±2.4 | 88.2±1.7 | 86.8±3.7/78.5±1.8 | 93.4±1.4 | *57.4*±0.7/51.7±1.3 | 78.5±2.8 | 55.7±1.8/51.4±1.2 | 78.5±2.8 | 41.1±1.3/53.0±1.3 | 78.5±2.8 | 78.7±1.3/78.7±0.7 | 80.5±1.2 | 63.1±0.9/53.3±0.7 | 89.8±1.4 |
| | GCN-BNN | 73.1±9.5/76.8±9.1 | 84.3±9.1 | 85.7±3.9/87.0±4.1 | 93.4±1.9 | 56.1±0.7/54.5±1.2 | 78.3±2.5 | 54.2±1.4/54.2±1.6 | 78.3±2.5 | 41.7±1.5/55.6±1.4 | 78.3±2.5 | 76.2±2.2/50.7±6.0 | 77.7±3.2 | 61.4±1.5/**65.1**±1.9 | 90.8±1.4 |
| | GCN-Ens | 74.0±6.1/78.0±4.9 | 88.9±1.9 | 86.9±4.4/82.4±4.0 | 94.2±1.1 | 56.8±0.6/47.6±1.5 | 80.4±2.5 | 55.3±1.7/52.2±2.0 | 80.4±2.5 | 41.2±1.2/57.1±1.3 | 80.4±2.5 | 81.0±1.0/62.5±2.6 | 81.2±1.3 | 61.9±0.8/63.2±1.4 | 91.3±1.2 |
| | GPN | 80.2±5.5/75.3±4.0 | 86.4±1.9 | 82.6±5.6/82.8±2.6 | 94.3±1.1 | 56.2±1.1/55.7±1.4 | 74.4±14.7 | *57.1*±3.8/56.5±1.3 | 74.4±14.7 | 58.4±0.9/*58.7*±3.3 | 74.4±14.7 | 52.1±0.8/76.8±1.7 | 78.5±2.2 | 62.4±3.5/49.1±2.0 | 92.4±1.2 |
| | SGCN | 47.9±11.1/61.7±6.4 | 39.5±27.2 | 59.5±15.0/59.2±17.4 | 59.1±26.4 | 49.3±0.6/50.6±0.8 | 11.7±12.2 | 48.5±0.8/51.4±0.9 | 11.7±12.2 | 47.8±2.4/46.0±2.0 | 11.7±12.2 | **93.8**±0.8/9.2±13.8 | 20.0±14.3 | 43.4±0.6/56.7±0.2 | 10.2±9.5 |
| | GCN-EBM | 74.3±6.1/75.9±5.1 | 88.8±1.9 | 86.2±3.8/85.5±3.6 | 94.1±1.2 | 56.7±0.7/56.6±0.7 | 80.0±2.6 | 55.3±1.8/54.9±2.1 | 80.0±2.6 | 41.2±1.3/40.1±1.1 | 80.0±2.6 | 80.8±1.0/*85.3*±1.4 | 81.0±1.7 | 61.9±0.8/61.5±0.9 | 91.1±1.2 |
| | GCN-HEAT | 74.3±6.1/75.6±6.0 | 88.8±1.9 | 86.7±4.4/84.9±4.5 | 94.1±1.2 | 56.7±0.7/55.9±0.9 | 80.0±2.6 | 55.3±1.8/54.9±2.8 | 80.0±2.6 | 41.2±1.3/58.3±2.0 | 80.0±2.6 | 80.8±1.0/79.0±4.3 | 81.0±1.7 | 61.9±0.8/63.6±1.6 | 91.1±1.2 |
| | GCNSafe | 74.3±6.1/**87.0**±2.6 | 88.8±1.9 | 86.7±4.3/83.3±3.0 | 94.1±1.2 | 56.7±0.7/51.9±1.4 | 80.0±2.6 | 55.3±1.8/51.8±1.9 | 80.0±2.6 | 41.2±1.3/42.9±1.2 | 80.0±2.6 | 80.8±1.0/58.7±0.8 | 81.0±1.7 | 61.9±0.8/55.6±1.3 | 91.1±1.2 |
| | **GCN-GEBM** | 74.3±6.1/*81.8*±3.5 | 88.8±1.9 | 86.7±4.3/**87.9**±3.9 | 94.1±1.2 | 56.7±0.7/**85.2**±2.2 | 80.0±2.6 | 55.3±1.8/**75.4**±6.2 | 80.0±2.6 | 41.2±1.3/**94.1**±6.6 | 80.0±2.6 | 80.8±1.0/52.6±1.3 | 81.0±1.7 | 61.9±0.8/*64.4*±1.5 | 91.1±1.2 |
| Amazon Photo | GCN-MCD | 74.6±4.3/50.0±1.1 | 92.0±1.1 | 81.7±2.8/49.9±2.2 | 95.3±2.0 | 57.3±0.8/49.8±1.5 | 89.8±1.1 | 52.0±1.7/49.7±1.5 | 89.8±1.1 | 42.8±1.1/49.5±1.4 | 89.8±1.1 | 83.5±1.1/49.9±1.4 | 91.9±0.5 | 62.0±0.6/49.6±1.5 | 97.4±0.2 |
| | GCN-DE | 76.4±3.3/61.2±2.5 | 91.7±1.1 | 82.0±2.4/73.4±2.5 | 94.6±2.0 | *58.4*±0.8/53.1±0.9 | 88.5±1.2 | 52.8±1.8/51.4±1.3 | 88.5±1.2 | 41.8±1.1/51.5±1.5 | 88.5±1.2 | 81.5±1.3/82.4±0.6 | 91.3±0.7 | 62.7±0.7/52.0±1.0 | 96.6±0.4 |
| | GCN-MCD+DE | 76.4±3.4/61.1±2.8 | 91.4±1.1 | 81.5±2.3/72.9±2.2 | 94.8±1.9 | 58.0±0.7/52.7±0.8 | 88.3±1.8 | 52.5±2.2/51.1±1.1 | 88.3±1.8 | 42.4±1.1/52.2±1.2 | 88.3±1.8 | 81.6±1.3/82.4±0.8 | 91.6±0.7 | 62.6±0.6/52.1±0.9 | 96.6±0.4 |
| | GCN-BNN | 71.9±5.2/74.6±5.8 | 91.6±2.0 | 85.5±3.8/*88.6*±4.0 | 95.7±1.9 | 56.1±1.0/54.4±1.4 | 89.3±2.1 | 50.2±2.2/49.2±2.0 | 89.3±2.1 | 43.9±0.9/56.6±1.9 | 89.3±2.1 | 76.9±2.6/67.6±4.8 | 90.4±2.7 | 63.7±2.4/*66.4*±3.0 | 97.6±0.4 |
| | GCN-Ens | 74.7±4.3/80.3±6.4 | 91.9±1.0 | *81.7*±2.8/83.5±6.5 | 95.3±1.9 | 57.3±0.8/51.4±1.2 | 89.9±1.0 | 52.3±1.8/51.3±1.2 | 89.9±1.0 | 42.6±1.1/55.0±1.3 | 89.9±1.0 | 83.6±1.1/76.9±1.6 | 92.1±0.5 | 62.0±0.6/65.2±1.2 | 97.5±0.2 |
| | GPN | 74.5±3.8/**87.3**±2.0 | 88.8±3.1 | **89.2**±3.1/76.2±4.1 | 93.9±2.2 | 55.4±1.1/54.8±1.0 | 86.5±1.8 | *55.3*±1.8/54.9±1.0 | 86.5±1.8 | 54.2±1.1/55.5±1.2 | 86.5±1.8 | 51.9±0.6/75.7±3.0 | 88.6±2.0 | 65.9±2.5/47.0±3.2 | 97.1±0.5 |
| | SGCN | 77.1±2.2/78.3±3.5 | 92.1±0.9 | 77.7±2.3/78.0±4.4 | 94.3±2.2 | 49.2±1.2/50.7±1.2 | 15.4±10.0 | 48.5±1.1/51.2±1.6 | 15.4±10.0 | 46.8±2.2/46.5±2.6 | 15.4±10.0 | **93.2**±1.5/23.9±33.6 | 12.2±8.6 | 43.3±1.1/56.4±3.1 | 15.0±5.9 |
| | GCN-EBM | 74.6±4.3/75.2±5.0 | 91.8±1.0 | 81.7±2.8/78.1±3.2 | 95.3±2.0 | 57.3±0.8/57.8±0.6 | 89.8±1.0 | 52.1±1.8/52.1±1.7 | 89.8±1.0 | 44.2±1.2/42.6±1.3 | 89.8±1.0 | 83.6±1.1/*90.3*±0.9 | 91.9±0.6 | 61.9±0.6/60.0±0.8 | 97.4±0.3 |
| | GCN-HEAT | 74.6±4.4/77.3±5.1 | 91.8±1.0 | 81.7±2.8/75.4±5.1 | 95.3±2.0 | 57.3±0.8/56.9±1.4 | 89.8±1.0 | 52.1±1.8/51.8±2.1 | 89.8±1.0 | 44.2±1.2/*57.3*±1.9 | 89.8±1.0 | 83.6±1.1/80.5±3.7 | 91.9±0.6 | 61.9±0.6/62.6±1.7 | 97.4±0.3 |
| | GCNSafe | 74.6±4.4/75.6±7.4 | 91.8±1.0 | 81.7±2.8/84.4±3.9 | 95.3±2.0 | 57.3±0.8/53.9±1.0 | 89.8±1.0 | 52.1±1.8/50.9±1.7 | 89.8±1.0 | 44.2±1.2/46.4±1.0 | 89.8±1.0 | 83.6±1.1/62.1±1.0 | 91.9±0.6 | 61.9±0.6/60.5±0.8 | 97.4±0.3 |
| | **GCN-GEBM** | 74.6±4.4/*83.6*±5.2 | 91.8±1.0 | 81.7±2.8/87.5±2.9 | 95.3±2.0 | 57.3±0.8/**84.8**±3.1 | 89.8±1.0 | 52.1±1.8/**64.2**±8.6 | 89.8±1.0 | 44.2±1.2/**92.4**±2.8 | 89.8±1.0 | 83.6±1.1/58.0±0.7 | 91.9±0.6 | 61.9±0.6/**68.3**±1.1 | 97.4±0.3 |
| Coauthor-CS | GCN-MCD | 89.6±1.5/50.1±1.1 | 92.3±0.8 | 93.7±0.8/49.9±1.7 | 95.1±0.5 | 68.6±0.6/50.1±0.9 | 89.9±1.0 | 69.6±1.1/50.1±0.8 | 89.9±1.0 | 34.6±0.5/50.1±0.8 | 89.9±1.0 | 76.1±0.7/50.2±0.7 | 90.8±0.4 | 63.8±0.3/50.3±1.0 | 97.9±0.4 |
| | GCN-DE | 89.2±1.3/73.7±1.4 | 91.3±0.7 | 93.1±0.8/83.1±0.9 | 94.4±0.7 | 68.3±0.9/59.4±0.8 | 88.8±1.1 | 64.3±1.2/67.9±1.1 | 88.8±1.1 | 33.7±0.7/60.3±0.9 | 88.8±1.1 | 70.9±0.9/76.8±0.8 | 89.6±0.6 | 62.9±0.4/55.7±0.6 | 97.1±0.5 |
| | GCN-MCD+DE | 89.0±1.7/73.6±1.4 | 91.4±0.5 | 92.8±1.1/82.8±1.1 | 94.3±0.6 | 68.6±0.8/59.6±0.8 | 88.5±1.0 | 64.6±1.1/68.1±1.0 | 88.5±1.0 | 33.8±0.7/59.6±0.9 | 88.5±1.0 | 70.6±0.9/76.5±0.9 | 89.6±0.6 | 63.0±0.6/55.9±0.6 | 97.1±0.4 |
| | GCN-BNN | 85.6±2.6/85.0±2.2 | 90.8±0.8 | 91.3±1.7/90.3±1.6 | 94.7±0.5 | 67.2±1.1/67.7±1.1 | 89.0±0.8 | 57.0±2.2/*73.7*±1.6 | 89.0±0.8 | 28.7±1.7/55.5±1.1 | 89.0±0.8 | 69.0±1.6/61.7±1.2 | 89.8±0.6 | 69.1±0.7/**71.8**±0.9 | 97.9±0.4 |
| | GCN-Ens | 89.7±1.4/87.3±2.4 | 92.4±0.8 | 93.7±0.9/92.3±1.1 | 95.2±0.5 | 68.6±1.0/60.5±2.2 | 90.0±0.9 | 69.6±1.1/67.8±2.2 | 90.0±0.9 | 32.8±1.1/60.7±0.6 | 90.0±0.9 | 76.0±0.7/75.0±0.7 | 90.8±0.4 | 63.9±0.2/69.1±0.6 | 97.9±0.4 |
| | GPN | 73.7±3.8/88.3±1.4 | 86.6±1.4 | 82.1±3.3/77.5±4.2 | 91.6±1.2 | 56.8±1.4/58.2±1.4 | 81.7±2.3 | 54.1±0.9/59.7±2.0 | 81.7±2.3 | 53.6±0.9/59.9±2.0 | 81.7±2.3 | 49.6±0.6/77.3±1.6 | 83.4±1.6 | 61.3±3.3/45.5±1.3 | 97.9±0.4 |
| | SGCN | 62.2±6.0/42.1±2.5 | 30.6±14.7 | 69.8±11.4/29.2±4.9 | 31.2±16.2 | 48.0±0.4/52.2±0.5 | 11.0±7.0 | 34.1±1.0/66.4±1.1 | 11.0±7.0 | 33.7±1.1/46.2±3.3 | 11.0±7.0 | **83.8**±3.3/15.0±2.5 | 12.1±9.1 | 39.0±2.3/63.2±1.1 | 14.1±6.5 |
| | GCN-EBM | 89.7±1.4/89.9±1.6 | 92.3±0.9 | 93.6±0.8/94.2±1.1 | 95.1±0.5 | 68.6±1.0/*69.6*±0.8 | 89.8±0.9 | 69.3±1.1/71.6±1.6 | 89.8±0.9 | 34.7±0.6/28.8±0.7 | 89.8±0.9 | 76.1±0.8/*81.9*±0.8 | 90.9±0.3 | 63.8±0.3/60.5±0.7 | 97.8±0.4 |
| | GCN-HEAT | 89.7±1.4/88.6±1.0 | 92.3±0.9 | 93.6±0.8/91.2±0.9 | 95.1±0.5 | 68.6±1.0/68.9±1.2 | 89.8±0.9 | 69.3±1.1/70.0±1.5 | 89.8±0.9 | 34.7±0.6/*69.8*±0.6 | 89.8±0.9 | 76.1±0.8/71.2±2.6 | 90.9±0.3 | 63.8±0.3/66.6±1.4 | 97.8±0.4 |
| | GCNSafe | 89.7±1.4/*92.1*±1.9 | 92.3±0.9 | 93.6±0.8/**96.6**±0.6 | 95.1±0.5 | 68.6±1.0/62.2±0.9 | 89.8±0.9 | 69.3±1.1/63.0±1.3 | 89.8±0.9 | 34.7±0.6/36.9±0.7 | 89.8±0.9 | 76.1±0.8/61.4±0.6 | 90.9±0.3 | 63.8±0.3/62.4±0.7 | 97.8±0.4 |
| | **GCN-GEBM** | 89.7±1.4/**92.8**±1.5 | 92.3±0.9 | 93.6±0.8/*96.1*±1.3 | 95.1±0.5 | 68.6±1.0/**95.6**±1.1 | 89.8±0.9 | 69.3±1.1/**99.5**±0.2 | 89.8±0.9 | 34.7±0.6/**90.0**±5.4 | 89.8±0.9 | 76.1±0.8/56.5±0.9 | 90.9±0.3 | 63.8±0.3/*69.5*±1.0 | 97.8±0.4 |
| Coauthor-Physics | GCN-MCD | 92.7±1.4/49.7±1.1 | 96.7±0.5 | 96.4±0.8/50.1±0.9 | 98.3±0.2 | 59.4±1.4/50.1±0.7 | 91.9±1.2 | 62.4±0.8/49.9±0.8 | 91.9±1.2 | 42.4±0.8/49.9±0.5 | 91.9±1.2 | 78.8±1.3/49.6±0.6 | 93.4±1.0 | 58.0±0.8/50.1±0.7 | 98.0±0.2 |
| | GCN-DE | 92.4±1.4/85.3±2.0 | 96.4±0.7 | 95.9±0.9/91.4±0.6 | 98.1±0.2 | 59.7±1.1/55.1±0.6 | 91.1±1.2 | 62.7±1.1/58.8±0.6 | 91.1±1.2 | 41.9±0.8/50.4±0.7 | 91.1±1.2 | 74.6±1.1/77.8±0.9 | 93.1±1.0 | 58.1±0.6/55.2±0.3 | 97.7±0.2 |
| | GCN-MCD+DE | 92.8±1.4/85.7±1.5 | 96.5±0.6 | 96.0±0.8/91.6±0.7 | 98.2±0.2 | 59.6±1.1/55.1±0.6 | 91.3±1.5 | 62.6±1.1/58.5±0.6 | 91.3±1.5 | 41.7±1.1/50.4±0.8 | 91.3±1.5 | 74.8±1.0/78.0±0.6 | 93.1±0.9 | 58.1±0.6/55.2±0.4 | 97.7±0.2 |
| | GCN-BNN | 89.3±3.7/87.5±6.0 | 96.6±1.1 | 94.4±1.1/94.5±1.8 | 98.1±0.3 | 59.7±1.0/60.0±1.5 | 91.7±1.4 | 56.5±2.8/*65.8*±3.1 | 91.7±1.4 | 35.7±0.8/55.5±2.0 | 91.7±1.4 | 74.3±1.5/68.0±2.3 | 93.4±0.9 | 61.5±1.3/**62.8**±1.7 | 98.4±0.2 |
| | GCN-Ens | 92.8±1.5/87.8±3.3 | 96.8±0.5 | 96.5±0.9/93.6±1.2 | 98.3±0.2 | 59.5±1.1/55.7±1.5 | 92.0±1.2 | 62.3±0.8/58.5±1.1 | 92.0±1.2 | 41.7±1.1/53.0±1.0 | 92.0±1.2 | 78.9±1.3/74.3±2.5 | 93.5±1.0 | 58.1±0.8/*62.2*±0.9 | 98.0±0.2 |
| | GPN | 84.8±3.0/90.3±2.8 | 96.8±0.4 | 83.7±2.0/93.4±1.3 | 97.7±0.3 | 59.6±1.2/58.7±1.1 | 90.3±1.4 | 56.4±0.9/60.6±1.3 | 90.3±1.4 | 55.7±0.7/*60.8*±1.3 | 90.3±1.4 | 53.0±0.8/78.2±1.5 | 92.3±1.0 | 61.7±5.4/48.5±2.7 | 98.8±0.2 |
| | SGCN | 92.4±1.3/92.1±2.6 | 96.9±0.4 | 96.3±0.9/96.0±1.4 | 98.3±0.2 | 60.5±0.6/*61.8*±1.7 | 91.3±1.4 | 62.1±0.9/61.9±2.6 | 91.3±1.4 | 41.6±0.4/40.1±0.8 | 91.3±1.4 | 78.2±1.5/*79.3*±2.5 | 93.3±0.7 | 58.0±0.9/56.9±1.5 | 97.9±0.6 |
| | GCN-EBM | 92.7±1.5/93.4±1.6 | 96.8±0.5 | 96.4±0.9/97.0±0.6 | 98.3±0.2 | 59.3±1.1/59.4±1.3 | 92.0±1.2 | 62.2±0.9/62.4±1.4 | 92.0±1.2 | 42.6±1.1/39.5±1.0 | 92.0±1.2 | 78.8±1.3/**82.0**±1.6 | 93.4±1.0 | 58.0±0.8/56.5±0.6 | 98.0±0.2 |
| | GCN-HEAT | 92.7±1.5/91.4±1.4 | 96.8±0.5 | 96.4±0.9/94.7±0.7 | 98.3±0.2 | 59.3±1.1/58.2±1.2 | 92.0±1.2 | 62.2±0.9/62.8±1.3 | 92.0±1.2 | 42.6±1.1/60.2±0.3 | 92.0±1.2 | 78.8±1.3/74.8±2.3 | 93.4±1.0 | 58.0±0.8/58.9±0.7 | 98.0±0.2 |
| | GCNSafe | 92.7±1.5/**95.8**±1.1 | 96.8±0.5 | 96.4±0.9/*98.1*±0.4 | 98.3±0.2 | 59.3±1.1/55.1±2.2 | 92.0±1.2 | 62.2±0.9/57.1±0.9 | 92.0±1.2 | 42.6±1.1/44.1±0.9 | 92.0±1.2 | 78.8±1.3/65.5±1.3 | 93.4±1.0 | 58.0±0.8/57.9±0.5 | 98.0±0.2 |
| | **GCN-GEBM** | 92.7±1.5/*94.9*±1.8 | 96.8±0.5 | 96.4±0.9/**98.4**±0.3 | 98.3±0.2 | 59.3±1.1/**81.8**±2.5 | 92.0±1.2 | 62.2±0.9/**95.3**±1.5 | 92.0±1.2 | 42.6±1.1/**72.7**±5.8 | 92.0±1.2 | 78.8±1.3/60.1±1.1 | 93.4±1.0 | 58.0±0.8/61.3±0.8 | 98.0±0.2 |

Table 6: Out-of-Distribution detection AUC-ROC (↑) using aleatoric or epistemic uncertainty (best and runner-up) in an inductive setting.

GEBM in Table 15. On all datasets, GEBM improves the AUC-ROC scores by 5.8 to 10.9 percentage points on average (16%-32% relative improvement) over the second-best ranked estimator.

### C.3 Misclassification Detection

We report AUC-ROC and AUC-PR for misclassification detection in both an inductive and transductive setting with the corresponding standard deviations in Tables 16 to 19. In alignment with previous work [64], we observe that often aleatoric uncertainty is more suitable for misclassification detection than epistemic uncertainty. Overall, GEBM performs similar to other epistemic measures. We again want to point out that the scope of this work does not encompass improvements on aleatoric uncertainty. The results on misclassification detection obtained in our work and previous studies indicate that this problem may be more suitable as a benchmark for aleatoric uncertainty.

### C.4 Calibration

Our framework, GEBM, is limited toward epistemic uncertainty estimation and leaves the aleatoric estimates and the classifier calibration unchanged. For completeness, we nonetheless report calibration

| Model | CoraML | CoraML LLM | Citeseer | PubMed | Amazon Computers | Amazon Photo | Coauthor CS | Coauthor Physics |
|---|---|---|---|---|---|---|---|---|
| GCN-MCD | 10.7/20.1 | 10.1/19.9 | 9.8/19.3 | 9.2/18.8 | 10.2/19.6 | 10.2/19.7 | 10.0/19.3 | 10.5/20.4 |
| GCN-DE | 6.3/14.2 | 8.0/16.9 | 7.8/16.2 | 7.3/13.6 | 7.2/15.9 | 7.1/15.8 | 6.5/13.2 | 8.1/16.8 |
| GCN-MCD+DE | 6.9/14.8 | 7.2/15.9 | 6.6/14.7 | 7.3/13.6 | 7.7/16.3 | 7.4/16.2 | 6.8/13.5 | 7.8/16.4 |
| GCN-BNN | 5.2/11.4 | 6.0/13.4 | 5.4/11.0 | 5.3/10.9 | 4.3/9.0 | 5.1/9.8 | 4.9/10.5 | 4.8/9.9 |
| GCN-Ens | 7.8/16.1 | 6.3/15.0 | 6.4/12.9 | 6.2/12.4 | 5.7/11.6 | 4.7/**9.0** | 4.8/10.4 | 6.2/13.3 |
| GPN | 7.0/15.2 | 5.4/11.2 | 6.6/12.9 | 5.8/12.4 | 5.7/11.8 | 5.3/11.7 | 7.2/13.9 | 6.1/14.5 |
| SGCN | 5.2/9.5 | 6.1/14.6 | 6.0/12.6 | **5.0**/11.7 | 9.1/18.5 | 7.6/15.1 | 9.3/17.9 | 4.6/11.2 |
| GCN-EBM | 4.5/**7.9** | **4.0**/8.2 | 5.2/10.3 | 5.2/12.0 | 4.2/10.7 | 4.9/10.9 | **4.1**/**7.1** | **4.4**/**8.4** |
| GCN-HEAT | **4.4**/10.4 | 5.4/12.9 | **3.8**/**9.2** | 5.6/12.8 | **3.5**/**8.6** | **4.5**/10.2 | 4.2/8.3 | 4.7/10.4 |
| GCNSafe | 5.3/9.1 | 4.4/**8.1** | 5.2/9.5 | 5.4/**8.9** | 5.9/13.2 | 6.4/12.9 | 5.5/10.7 | 5.8/11.7 |
| **GCN-GEBM** | **2.7 / 4.5** | **2.9 / 4.9** | **3.3 / 5.7** | **3.8 / 7.4** | **2.5 / 4.2** | **2.7 / 4.5** | **2.7 / 4.3** | **3.0 / 5.0** |

Table 7: Average o.o.d. detection rank (AUC-ROC) (↓) of epistemic uncertainty versus other epistemic measures / all uncertainty measures over all distribution shifts in an inductive setting (best and runner-up).

for the GCN used at the backbone of our experiments. Note that since our model does not change the output of the classifier, the calibration could be further improved using post hoc methods like temperature scaling [26]. This, however, is beyond the scope of this work.

**Expected Calibration Error (ECE).** The expected calibration error (↓) [50] measures how well the predicted probabilities that are normalized to the interval $[0, 1]$ match the true predictive accuracy. To that end, we bin each prediction into $B = 20$ bins according to their confidence (i.e. maximum softmax response $\max_c \boldsymbol{p}(y = c \mid \boldsymbol{x})$. For each bin, we then compute the average accuracy and compute the ECE as the mean over all bins weighted by their size.

$$\text{ECE} = \sum_k \frac{|B_k|}{n} |\text{accuracy}(B_k) - \text{confidence}(B_k)|$$

**Brier Score.** A similar metric is the Brier score [7] (↓) which computes the mean squared distance between the predicted probabilities and the one-hot encoded true labels.

$$\text{Brier} = \frac{1}{n} \sum_i \|\boldsymbol{p}(\boldsymbol{y} \mid \boldsymbol{x}_i) - \boldsymbol{y}_i\|_2^2$$

We report ECE and Brier scores for inductive and transductive settings in Tables 20 to 23. In terms of calibration metrics, none of the considered approaches consistently shows strong merits. We remark that work on GEBM and GNN calibration is somewhat orthogonal and our framework can be applied to any well-calibrated GNN backbone.

### C.5 Backbone Architecture

We report AUC-ROC and AUC-PR for o.o.d. detection using GEBM and different GNN backbones with corresponding standard deviations in Tables 24 and 25. GEBM is effective on all models and achieves the highest scores on most distribution shifts.

### C.6 Bias of Diffusion Operators

At the core of many GNNs and also our GEBM framework lies a diffusion operator $P_{\boldsymbol{A}} : \mathbb{R}^n \to \mathbb{R}^n$. Here, we discuss three typical realizations and the bias the introduce.

**Symmetric Diffusion.** Symmetric diffusion normalizes the adjacency matrix as $\hat{\boldsymbol{A}} = \boldsymbol{D}^{-1/2}\boldsymbol{A}\boldsymbol{D}^{-1/2}$. Here, $\boldsymbol{D} = \text{diag}(\text{deg}(v_1), \cdots \text{deg}(v_n))$ is a diagonal matrix of node degrees. The symmetric diffusion operator has a dominant eigenvector that correlates with the node degree $\boldsymbol{v}_{\max} \propto \boldsymbol{d}^{1/2}$ and therefore also centrality [12]. Repeated application of this diffusion process to any arbitrary signal will concentrate confidence at high degree nodes. Applying this diffusion to a confidence measure like the logits of GNN will therefore always favor high degree nodes.

| Dataset | Model | LoC (*last*) AUC-ROC↑ (Alea. / Epi.) | LoC (*last*) Acc.↑ | LoC (*hetero.*) AUC-ROC↑ (Alea. / Epi.) | LoC (*hetero.*) Acc.↑ | Ber($\hat{p}$) (*near*) AUC-ROC↑ (Alea. / Epi.) | Ber($\hat{p}$) (*near*) Acc.↑ | Ber(0.5) AUC-ROC↑ (Alea. / Epi.) | Ber(0.5) Acc.↑ | $\mathcal{N}(0,1)$ (*far*) AUC-ROC↑ (Alea. / Epi.) | $\mathcal{N}(0,1)$ (*far*) Acc.↑ | Page Rank AUC-ROC↑ (Alea. / Epi.) | Page Rank Acc.↑ | Homophily AUC-ROC↑ (Alea. / Epi.) | Homophily Acc.↑ |
|---|---|---|---|---|---|---|---|---|---|---|---|---|---|---|---|
| CoraML | GCN-MCD | $87.5_{\pm1.6}/50.3_{\pm2.7}$ | $90.4_{\pm0.9}$ | $84.9_{\pm1.7}/50.2_{\pm1.7}$ | $85.2_{\pm3.3}$ | $67.5_{\pm1.4}/50.3_{\pm2.8}$ | $82.0_{\pm0.8}$ | $61.8_{\pm3.2}/50.0_{\pm1.9}$ | $82.0_{\pm0.8}$ | $31.7_{\pm2.3}/49.9_{\pm2.6}$ | $82.0_{\pm0.8}$ | $71.4_{\pm0.8}/50.4_{\pm2.6}$ | $82.6_{\pm1.5}$ | $69.0_{\pm1.7}/48.9_{\pm2.3}$ | $93.9_{\pm0.8}$ |
| | GCN-DE | $84.5_{\pm1.9}/71.9_{\pm1.9}$ | $87.3_{\pm1.2}$ | $81.9_{\pm2.5}/67.0_{\pm3.4}$ | $80.9_{\pm2.3}$ | $66.7_{\pm1.6}/57.6_{\pm2.8}$ | $78.4_{\pm1.9}$ | $52.2_{\pm3.1}/70.6_{\pm2.2}$ | $78.4_{\pm1.9}$ | $27.2_{\pm1.8}/55.6_{\pm3.1}$ | $78.4_{\pm1.9}$ | $64.3_{\pm1.2}/68.8_{\pm1.4}$ | $79.0_{\pm2.3}$ | $67.5_{\pm2.2}/57.1_{\pm1.6}$ | $90.6_{\pm1.8}$ |
| | GCN-MCD+DE | $84.6_{\pm1.7}/72.7_{\pm1.9}$ | $87.6_{\pm1.2}$ | $80.9_{\pm2.9}/66.7_{\pm3.2}$ | $80.1_{\pm2.9}$ | $66.8_{\pm1.9}/58.0_{\pm2.6}$ | $78.1_{\pm2.2}$ | $52.9_{\pm3.4}/71.1_{\pm2.7}$ | $78.1_{\pm2.2}$ | $27.9_{\pm1.8}/57.6_{\pm2.3}$ | $78.1_{\pm2.2}$ | $64.7_{\pm1.5}/68.4_{\pm1.6}$ | $79.5_{\pm1.7}$ | $67.8_{\pm2.1}/56.9_{\pm2.0}$ | $90.8_{\pm1.7}$ |
| | GCN-BNN | $84.7_{\pm2.3}/84.7_{\pm2.2}$ | $89.0_{\pm1.2}$ | $78.6_{\pm3.3}/77.6_{\pm3.6}$ | $82.1_{\pm3.4}$ | $65.8_{\pm1.2}/63.4_{\pm1.4}$ | $80.6_{\pm2.0}$ | $53.5_{\pm5.3}/70.2_{\pm5.2}$ | $80.6_{\pm2.0}$ | $32.8_{\pm2.6}/54.0_{\pm1.7}$ | $80.6_{\pm2.0}$ | $67.6_{\pm0.9}/58.1_{\pm1.4}$ | $80.4_{\pm2.3}$ | $69.5_{\pm2.1}/70.8_{\pm2.3}$ | $93.1_{\pm1.5}$ |
| | GCN-Ens | $87.6_{\pm1.6}/74.6_{\pm3.0}$ | $90.3_{\pm0.9}$ | $84.9_{\pm1.8}/61.7_{\pm6.7}$ | $85.2_{\pm3.3}$ | $67.7_{\pm1.5}/51.6_{\pm2.7}$ | $81.8_{\pm0.8}$ | $61.4_{\pm2.4}/66.4_{\pm3.5}$ | $81.8_{\pm0.8}$ | $32.3_{\pm1.5}/52.6_{\pm2.3}$ | $81.8_{\pm0.8}$ | $71.5_{\pm0.7}/59.8_{\pm3.1}$ | $82.7_{\pm1.5}$ | $68.9_{\pm1.7}/70.4_{\pm2.5}$ | $94.0_{\pm0.5}$ |
| | GPN | $84.7_{\pm0.9}/88.1_{\pm1.6}$ | $90.0_{\pm1.0}$ | $82.2_{\pm4.2}/83.9_{\pm1.8}$ | $85.5_{\pm2.2}$ | $52.9_{\pm2.9}/53.8_{\pm3.4}$ | $78.8_{\pm2.1}$ | $52.4_{\pm2.7}/52.8_{\pm3.1}$ | $78.8_{\pm2.1}$ | $50.6_{\pm2.4}/55.0_{\pm3.7}$ | $78.8_{\pm2.1}$ | $52.2_{\pm0.9}/65.2_{\pm1.9}$ | $78.4_{\pm1.8}$ | $74.2_{\pm2.7}/57.3_{\pm2.5}$ | $91.1_{\pm1.8}$ |
| | SGCN | $87.7_{\pm1.3}/89.1_{\pm1.7}$ | $91.0_{\pm1.2}$ | $86.6_{\pm2.3}/89.1_{\pm2.1}$ | $88.5_{\pm2.0}$ | $57.5_{\pm9.0}/59.1_{\pm7.2}$ | $51.7_{\pm30.2}$ | $44.8_{\pm15.5}/68.6_{\pm4.2}$ | $51.7_{\pm30.2}$ | $28.3_{\pm2.5}/35.8_{\pm11.4}$ | $51.7_{\pm30.2}$ | $71.0_{\pm1.4}/68.4_{\pm2.2}$ | $83.3_{\pm1.3}$ | $70.0_{\pm1.6}/71.2_{\pm1.6}$ | $94.2_{\pm1.0}$ |
| | GCN-EBM | $87.6_{\pm1.7}/87.5_{\pm1.8}$ | $90.4_{\pm0.9}$ | $84.8_{\pm1.9}/85.3_{\pm2.0}$ | $85.0_{\pm3.2}$ | $68.1_{\pm1.5}/68.7_{\pm1.5}$ | $81.9_{\pm0.8}$ | $62.3_{\pm3.0}/61.1_{\pm4.9}$ | $81.9_{\pm0.8}$ | $31.9_{\pm1.5}/24.7_{\pm1.6}$ | $81.9_{\pm0.8}$ | $71.4_{\pm0.8}/73.0_{\pm1.7}$ | $82.5_{\pm1.6}$ | $68.8_{\pm1.7}/67.9_{\pm2.0}$ | $94.1_{\pm0.6}$ |
| | GCN-HEAT | $87.6_{\pm1.7}/86.2_{\pm1.9}$ | $90.4_{\pm0.9}$ | $84.8_{\pm1.9}/82.4_{\pm3.1}$ | $85.0_{\pm3.2}$ | $68.1_{\pm1.5}/66.9_{\pm2.2}$ | $81.9_{\pm0.8}$ | $62.3_{\pm3.0}/65.9_{\pm4.3}$ | $81.9_{\pm0.8}$ | $31.9_{\pm1.5}/76.7_{\pm1.8}$ | $81.9_{\pm0.8}$ | $71.4_{\pm0.8}/70.2_{\pm2.1}$ | $82.5_{\pm1.6}$ | $68.8_{\pm1.7}/69.5_{\pm2.1}$ | $94.1_{\pm0.6}$ |
| | GCNSafe | $87.6_{\pm1.7}/90.0_{\pm1.6}$ | $90.4_{\pm0.9}$ | $84.8_{\pm1.9}/89.0_{\pm1.5}$ | $85.0_{\pm3.2}$ | $68.1_{\pm1.5}/60.2_{\pm1.5}$ | $81.9_{\pm0.8}$ | $62.3_{\pm3.0}/56.2_{\pm2.9}$ | $81.9_{\pm0.8}$ | $31.9_{\pm1.5}/33.8_{\pm1.4}$ | $81.9_{\pm0.8}$ | $71.4_{\pm0.8}/49.5_{\pm1.0}$ | $82.5_{\pm1.6}$ | $68.8_{\pm1.7}/70.3_{\pm1.9}$ | $94.1_{\pm0.6}$ |
| | **GCN-GEBM** | $87.6_{\pm1.7}/89.7_{\pm0.9}$ | $90.4_{\pm0.9}$ | $84.8_{\pm1.9}/87.9_{\pm1.4}$ | $85.0_{\pm3.2}$ | $68.1_{\pm1.5}/78.9_{\pm2.4}$ | $81.9_{\pm0.8}$ | $62.3_{\pm3.0}/92.4_{\pm3.7}$ | $81.9_{\pm0.8}$ | $31.9_{\pm1.5}/89.8_{\pm1.7}$ | $81.9_{\pm0.8}$ | $71.4_{\pm0.8}/52.7_{\pm1.0}$ | $82.5_{\pm1.6}$ | $68.8_{\pm1.7}/73.1_{\pm1.5}$ | $94.1_{\pm0.6}$ |
| CoraML-LLM | GCN-MCD | $88.3_{\pm2.2}/50.3_{\pm1.9}$ | $91.2_{\pm0.7}$ | $86.5_{\pm2.4}/50.5_{\pm2.4}$ | $90.6_{\pm2.0}$ | n.a./n.a. | n.a. | $69.3_{\pm1.3}/49.5_{\pm2.2}$ | $82.6_{\pm1.5}$ | $39.5_{\pm2.3}/50.0_{\pm2.3}$ | $82.6_{\pm1.5}$ | $70.1_{\pm1.5}/50.1_{\pm2.5}$ | $84.9_{\pm2.1}$ | $70.7_{\pm1.3}/50.0_{\pm2.6}$ | $94.1_{\pm1.1}$ |
| | GCN-DE | $86.8_{\pm2.1}/75.1_{\pm2.4}$ | $89.7_{\pm1.2}$ | $85.1_{\pm3.4}/73.1_{\pm3.2}$ | $88.9_{\pm1.7}$ | n.a./n.a. | n.a. | $74.6_{\pm1.2}/36.9_{\pm2.2}$ | $79.9_{\pm1.8}$ | $38.1_{\pm2.3}/62.5_{\pm2.8}$ | $79.9_{\pm1.8}$ | $63.9_{\pm1.5}/69.6_{\pm1.5}$ | $82.2_{\pm2.7}$ | $68.8_{\pm1.3}/61.3_{\pm1.5}$ | $91.7_{\pm1.4}$ |
| | GCN-MCD+DE | $87.0_{\pm1.5}/75.3_{\pm2.0}$ | $89.4_{\pm0.9}$ | $85.0_{\pm3.0}/72.6_{\pm3.6}$ | $88.9_{\pm1.7}$ | n.a./n.a. | n.a. | $74.7_{\pm1.8}/37.3_{\pm3.3}$ | $80.3_{\pm1.8}$ | $36.4_{\pm2.3}/61.3_{\pm2.4}$ | $80.3_{\pm1.8}$ | $64.0_{\pm1.4}/69.8_{\pm1.4}$ | $82.7_{\pm2.6}$ | $68.9_{\pm1.9}/61.4_{\pm1.9}$ | $91.4_{\pm1.5}$ |
| | GCN-BNN | $88.2_{\pm1.9}/84.0_{\pm2.4}$ | $91.4_{\pm0.8}$ | $86.0_{\pm2.8}/83.1_{\pm3.1}$ | $91.0_{\pm2.0}$ | n.a./n.a. | n.a. | $69.9_{\pm1.6}/41.2_{\pm3.7}$ | $82.8_{\pm1.6}$ | $38.7_{\pm1.0}/63.6_{\pm2.8}$ | $82.8_{\pm1.6}$ | $67.8_{\pm1.2}/53.9_{\pm3.0}$ | $84.0_{\pm2.2}$ | $72.9_{\pm1.8}/73.6_{\pm2.2}$ | $94.0_{\pm1.3}$ |
| | GCN-Ens | $88.3_{\pm2.1}/84.9_{\pm1.4}$ | $91.4_{\pm0.7}$ | $86.6_{\pm2.3}/77.3_{\pm5.0}$ | $90.6_{\pm2.3}$ | n.a./n.a. | n.a. | $69.6_{\pm2.0}/55.9_{\pm4.0}$ | $82.6_{\pm1.5}$ | $36.6_{\pm1.1}/63.6_{\pm2.9}$ | $82.6_{\pm1.5}$ | $70.1_{\pm1.4}/64.1_{\pm2.3}$ | $85.0_{\pm2.1}$ | $70.8_{\pm1.2}/70.2_{\pm1.8}$ | $94.0_{\pm1.1}$ |
| | GPN | $86.3_{\pm0.9}/90.3_{\pm1.0}$ | $90.5_{\pm1.1}$ | $85.6_{\pm4.1}/87.9_{\pm1.6}$ | $88.5_{\pm2.2}$ | n.a./n.a. | n.a. | $54.0_{\pm2.3}/54.7_{\pm2.3}$ | $79.7_{\pm1.8}$ | $52.2_{\pm2.2}/55.3_{\pm2.4}$ | $79.7_{\pm1.8}$ | $52.1_{\pm1.0}/64.4_{\pm1.3}$ | $80.9_{\pm2.2}$ | $76.0_{\pm1.6}/60.4_{\pm2.3}$ | $92.5_{\pm1.0}$ |
| | SGCN | $89.3_{\pm1.5}/90.3_{\pm1.2}$ | $90.3_{\pm0.9}$ | $87.7_{\pm2.5}/89.0_{\pm2.6}$ | $90.6_{\pm1.6}$ | n.a./n.a. | n.a. | $66.1_{\pm2.2}/55.7_{\pm12.3}$ | $68.7_{\pm21.0}$ | $35.0_{\pm2.9}/40.6_{\pm8.3}$ | $68.7_{\pm21.0}$ | $69.8_{\pm1.0}/70.3_{\pm2.0}$ | $82.0_{\pm2.7}$ | $69.3_{\pm1.3}/69.1_{\pm2.2}$ | $93.1_{\pm1.1}$ |
| | GCN-EBM | $88.2_{\pm2.2}/88.2_{\pm2.4}$ | $91.4_{\pm0.7}$ | $86.6_{\pm2.4}/87.0_{\pm2.3}$ | $90.7_{\pm2.0}$ | n.a./n.a. | n.a. | $69.5_{\pm1.7}/69.4_{\pm2.4}$ | $82.6_{\pm1.6}$ | $38.7_{\pm3.8}/34.1_{\pm3.4}$ | $82.6_{\pm1.6}$ | $70.1_{\pm1.4}/71.9_{\pm2.1}$ | $85.0_{\pm2.2}$ | $70.8_{\pm1.3}/69.0_{\pm1.5}$ | $93.8_{\pm1.0}$ |
| | GCN-HEAT | $88.2_{\pm2.2}/84.7_{\pm3.9}$ | $91.4_{\pm0.7}$ | $86.6_{\pm2.4}/83.8_{\pm5.0}$ | $90.7_{\pm2.0}$ | n.a./n.a. | n.a. | $69.5_{\pm1.7}/69.9_{\pm3.0}$ | $82.6_{\pm1.6}$ | $38.7_{\pm3.8}/66.3_{\pm4.2}$ | $82.6_{\pm1.6}$ | $70.1_{\pm1.4}/66.6_{\pm3.7}$ | $85.0_{\pm2.2}$ | $70.8_{\pm1.3}/69.8_{\pm1.9}$ | $93.8_{\pm1.0}$ |
| | GCNSafe | $88.2_{\pm2.2}/91.0_{\pm2.2}$ | $91.4_{\pm0.7}$ | $86.6_{\pm2.4}/91.7_{\pm1.4}$ | $90.7_{\pm2.0}$ | n.a./n.a. | n.a. | $69.5_{\pm1.7}/57.7_{\pm1.8}$ | $82.6_{\pm1.6}$ | $38.7_{\pm3.8}/42.4_{\pm3.4}$ | $82.6_{\pm1.6}$ | $70.1_{\pm1.4}/50.1_{\pm0.6}$ | $85.0_{\pm2.2}$ | $70.8_{\pm1.3}/70.7_{\pm1.9}$ | $93.8_{\pm1.0}$ |
| | **GCN-GEBM** | $88.2_{\pm2.2}/89.7_{\pm1.5}$ | $91.4_{\pm0.7}$ | $86.6_{\pm2.4}/89.1_{\pm2.6}$ | $90.7_{\pm2.0}$ | n.a./n.a. | n.a. | $69.5_{\pm1.7}/81.1_{\pm3.6}$ | $82.6_{\pm1.6}$ | $38.7_{\pm3.8}/86.0_{\pm2.5}$ | $82.6_{\pm1.6}$ | $70.1_{\pm1.4}/51.2_{\pm0.5}$ | $85.0_{\pm2.2}$ | $70.8_{\pm1.3}/72.9_{\pm1.6}$ | $93.8_{\pm1.0}$ |
| Citeseer | GCN-MCD | $85.2_{\pm1.3}/49.1_{\pm2.8}$ | $83.5_{\pm1.2}$ | $86.0_{\pm2.0}/49.6_{\pm2.0}$ | $91.1_{\pm0.8}$ | $66.9_{\pm1.0}/49.1_{\pm2.1}$ | $82.1_{\pm1.0}$ | $64.2_{\pm3.9}/50.2_{\pm2.0}$ | $82.1_{\pm1.0}$ | $32.0_{\pm1.0}/49.4_{\pm1.6}$ | $82.1_{\pm1.0}$ | $65.1_{\pm0.7}/50.1_{\pm1.9}$ | $82.9_{\pm1.1}$ | $51.0_{\pm0.9}/49.8_{\pm1.8}$ | $86.3_{\pm1.5}$ |
| | GCN-DE | $81.4_{\pm1.8}/70.3_{\pm2.6}$ | $80.1_{\pm1.7}$ | $81.7_{\pm2.3}/68.3_{\pm2.6}$ | $86.4_{\pm1.0}$ | $64.8_{\pm1.3}/58.5_{\pm1.8}$ | $76.8_{\pm1.5}$ | $49.9_{\pm4.2}/79.8_{\pm2.1}$ | $76.8_{\pm1.5}$ | $25.3_{\pm1.5}/60.8_{\pm1.9}$ | $76.8_{\pm1.5}$ | $59.0_{\pm1.2}/59.2_{\pm1.6}$ | $79.0_{\pm1.3}$ | $50.6_{\pm1.5}/50.0_{\pm1.3}$ | $81.6_{\pm1.6}$ |
| | GCN-MCD+DE | $81.7_{\pm1.8}/70.7_{\pm2.4}$ | $80.2_{\pm1.7}$ | $81.1_{\pm2.3}/69.2_{\pm2.2}$ | $86.2_{\pm1.1}$ | $64.7_{\pm1.4}/59.2_{\pm1.7}$ | $76.8_{\pm1.8}$ | $51.5_{\pm3.6}/80.6_{\pm1.8}$ | $76.8_{\pm1.8}$ | $25.3_{\pm1.4}/60.4_{\pm1.8}$ | $76.8_{\pm1.8}$ | $59.1_{\pm1.2}/59.4_{\pm1.4}$ | $79.2_{\pm1.0}$ | $50.7_{\pm1.1}/49.8_{\pm1.2}$ | $81.7_{\pm1.5}$ |
| | GCN-BNN | $83.8_{\pm1.8}/83.7_{\pm1.8}$ | $82.5_{\pm1.4}$ | $83.6_{\pm2.2}/83.4_{\pm2.4}$ | $90.2_{\pm0.7}$ | $68.0_{\pm1.0}/63.7_{\pm1.0}$ | $81.2_{\pm1.3}$ | $56.6_{\pm4.9}/74.3_{\pm3.7}$ | $81.2_{\pm1.3}$ | $29.5_{\pm2.1}/60.5_{\pm1.4}$ | $81.2_{\pm1.3}$ | $62.3_{\pm0.9}/55.0_{\pm1.1}$ | $81.3_{\pm1.1}$ | $51.5_{\pm1.0}/52.2_{\pm1.1}$ | $86.6_{\pm1.1}$ |
| | GCN-Ens | $85.4_{\pm0.9}/74.3_{\pm3.1}$ | $83.8_{\pm1.1}$ | $86.1_{\pm1.9}/77.9_{\pm3.1}$ | $91.2_{\pm0.8}$ | $67.0_{\pm1.2}/61.1_{\pm1.0}$ | $82.0_{\pm1.0}$ | $63.9_{\pm2.9}/82.6_{\pm2.0}$ | $82.0_{\pm1.0}$ | $31.3_{\pm2.1}/54.0_{\pm2.8}$ | $82.0_{\pm1.0}$ | $65.1_{\pm0.7}/51.3_{\pm1.4}$ | $82.6_{\pm0.8}$ | $51.0_{\pm0.9}/52.9_{\pm1.4}$ | $86.4_{\pm1.6}$ |
| | GPN | $84.2_{\pm3.1}/84.9_{\pm2.7}$ | $84.9_{\pm1.9}$ | $87.8_{\pm2.1}/83.0_{\pm3.0}$ | $90.9_{\pm0.9}$ | $60.2_{\pm1.7}/56.3_{\pm1.6}$ | $83.3_{\pm1.9}$ | $56.7_{\pm2.8}/57.8_{\pm3.3}$ | $83.3_{\pm1.9}$ | $52.2_{\pm1.0}/61.7_{\pm1.6}$ | $83.3_{\pm1.9}$ | $45.0_{\pm0.8}/50.7_{\pm1.6}$ | $82.1_{\pm1.5}$ | $53.4_{\pm0.6}/49.7_{\pm0.9}$ | $88.0_{\pm1.0}$ |
| | SGCN | $85.2_{\pm2.7}/88.4_{\pm2.9}$ | $83.5_{\pm1.9}$ | $86.7_{\pm1.9}/89.7_{\pm2.2}$ | $90.6_{\pm0.6}$ | $66.3_{\pm1.4}/63.5_{\pm1.7}$ | $81.5_{\pm1.8}$ | $65.6_{\pm4.7}/67.9_{\pm7.4}$ | $81.5_{\pm1.8}$ | $27.5_{\pm1.2}/24.0_{\pm1.2}$ | $81.5_{\pm1.8}$ | $64.0_{\pm1.1}/62.8_{\pm2.3}$ | $82.5_{\pm1.0}$ | $51.5_{\pm1.0}/51.6_{\pm1.2}$ | $85.9_{\pm1.4}$ |
| | GCN-EBM | $85.3_{\pm1.0}/85.7_{\pm1.2}$ | $83.6_{\pm1.1}$ | $85.8_{\pm1.9}/85.5_{\pm2.1}$ | $91.1_{\pm0.9}$ | $67.1_{\pm1.5}/65.8_{\pm2.0}$ | $82.0_{\pm1.0}$ | $63.6_{\pm4.6}/55.0_{\pm6.5}$ | $82.0_{\pm1.0}$ | $31.7_{\pm1.7}/23.3_{\pm1.2}$ | $82.0_{\pm1.0}$ | $65.1_{\pm0.8}/66.3_{\pm1.3}$ | $82.4_{\pm1.0}$ | $50.9_{\pm0.9}/50.6_{\pm1.0}$ | $86.4_{\pm1.5}$ |
| | GCN-HEAT | $85.3_{\pm1.0}/83.6_{\pm1.8}$ | $83.6_{\pm1.1}$ | $85.8_{\pm1.9}/83.5_{\pm2.2}$ | $91.1_{\pm0.9}$ | $67.1_{\pm1.5}/65.4_{\pm1.3}$ | $82.0_{\pm1.0}$ | $63.6_{\pm4.6}/82.1_{\pm2.2}$ | $82.0_{\pm1.0}$ | $31.7_{\pm1.7}/81.1_{\pm1.3}$ | $82.0_{\pm1.0}$ | $65.1_{\pm0.8}/61.7_{\pm1.1}$ | $82.4_{\pm1.0}$ | $50.9_{\pm0.9}/52.9_{\pm1.0}$ | $86.4_{\pm1.5}$ |
| | GCNSafe | $85.3_{\pm1.0}/88.9_{\pm1.4}$ | $83.6_{\pm1.1}$ | $85.8_{\pm1.9}/87.0_{\pm2.0}$ | $91.1_{\pm0.9}$ | $67.1_{\pm1.5}/59.3_{\pm2.0}$ | $82.0_{\pm1.0}$ | $63.6_{\pm4.6}/53.0_{\pm4.2}$ | $82.0_{\pm1.0}$ | $31.7_{\pm1.7}/32.1_{\pm1.2}$ | $82.0_{\pm1.0}$ | $65.1_{\pm0.8}/42.9_{\pm0.9}$ | $82.4_{\pm1.0}$ | $50.9_{\pm0.9}/53.2_{\pm0.7}$ | $86.4_{\pm1.5}$ |
| | **GCN-GEBM** | $85.3_{\pm1.0}/87.1_{\pm2.2}$ | $83.6_{\pm1.1}$ | $85.8_{\pm1.9}/86.7_{\pm2.2}$ | $91.1_{\pm0.9}$ | $67.1_{\pm1.5}/66.6_{\pm1.2}$ | $82.0_{\pm1.0}$ | $63.6_{\pm4.6}/98.5_{\pm0.8}$ | $82.0_{\pm1.0}$ | $31.7_{\pm1.7}/81.1_{\pm2.8}$ | $82.0_{\pm1.0}$ | $65.1_{\pm0.8}/48.0_{\pm1.1}$ | $82.4_{\pm1.0}$ | $50.9_{\pm0.9}/51.9_{\pm1.1}$ | $86.4_{\pm1.5}$ |
| PubMed | GCN-MCD | $66.0_{\pm3.0}/50.1_{\pm1.0}$ | $91.6_{\pm1.6}$ | $63.2_{\pm4.5}/50.2_{\pm1.2}$ | $83.7_{\pm1.5}$ | n.a./n.a. | n.a. | $65.9_{\pm7.0}/50.0_{\pm0.8}$ | $78.1_{\pm1.0}$ | $29.9_{\pm1.0}/49.7_{\pm1.0}$ | $78.1_{\pm1.0}$ | $67.8_{\pm1.2}/50.1_{\pm1.1}$ | $77.9_{\pm2.6}$ | $55.6_{\pm0.9}/50.1_{\pm0.9}$ | $88.9_{\pm2.8}$ |
| | GCN-DE | $65.1_{\pm2.7}/57.2_{\pm3.5}$ | $91.2_{\pm1.7}$ | $63.8_{\pm3.4}/58.6_{\pm2.6}$ | $80.5_{\pm2.1}$ | n.a./n.a. | n.a. | $62.6_{\pm7.0}/69.1_{\pm2.4}$ | $74.9_{\pm2.1}$ | $26.5_{\pm0.6}/45.8_{\pm1.7}$ | $74.9_{\pm2.1}$ | $59.2_{\pm1.1}/70.8_{\pm1.0}$ | $76.0_{\pm2.6}$ | $54.7_{\pm1.1}/48.9_{\pm1.0}$ | $84.8_{\pm2.3}$ |
| | GCN-MCD+DE | $65.8_{\pm3.0}/57.5_{\pm3.9}$ | $90.8_{\pm2.1}$ | $64.1_{\pm4.3}/59.1_{\pm2.7}$ | $80.7_{\pm1.8}$ | n.a./n.a. | n.a. | $62.1_{\pm5.9}/69.1_{\pm2.3}$ | $75.4_{\pm1.5}$ | $26.5_{\pm1.0}/46.0_{\pm1.6}$ | $75.4_{\pm1.5}$ | $59.1_{\pm1.3}/70.3_{\pm0.9}$ | $76.3_{\pm1.9}$ | $54.7_{\pm0.8}/49.1_{\pm1.0}$ | $84.9_{\pm1.8}$ |
| | GCN-BNN | $65.3_{\pm4.9}/64.5_{\pm5.2}$ | $90.7_{\pm1.6}$ | $63.1_{\pm6.1}/63.0_{\pm6.3}$ | $84.0_{\pm1.7}$ | n.a./n.a. | n.a. | $60.1_{\pm5.3}/66.9_{\pm5.9}$ | $78.2_{\pm1.6}$ | $32.7_{\pm2.2}/48.1_{\pm2.4}$ | $78.2_{\pm1.6}$ | $64.4_{\pm2.2}/51.0_{\pm3.7}$ | $76.1_{\pm4.2}$ | $57.0_{\pm1.4}/58.7_{\pm1.5}$ | $89.3_{\pm1.9}$ |
| | GCN-Ens | $66.0_{\pm2.9}/51.8_{\pm3.8}$ | $91.7_{\pm1.5}$ | $63.2_{\pm4.6}/49.1_{\pm5.8}$ | $83.8_{\pm1.4}$ | n.a./n.a. | n.a. | $65.6_{\pm6.8}/76.2_{\pm3.1}$ | $78.1_{\pm0.9}$ | $30.1_{\pm0.7}/51.9_{\pm2.3}$ | $78.1_{\pm0.9}$ | $67.9_{\pm1.1}/48.9_{\pm4.5}$ | $77.9_{\pm2.6}$ | $55.7_{\pm0.9}/56.3_{\pm2.2}$ | $88.9_{\pm2.8}$ |
| | GPN | $65.8_{\pm2.9}/64.0_{\pm5.9}$ | $92.3_{\pm0.8}$ | $62.7_{\pm4.9}/68.0_{\pm4.3}$ | $85.6_{\pm1.2}$ | n.a./n.a. | n.a. | $57.0_{\pm3.7}/56.1_{\pm3.4}$ | $79.8_{\pm1.1}$ | $54.6_{\pm2.2}/64.4_{\pm3.7}$ | $79.8_{\pm1.1}$ | $46.9_{\pm0.7}/70.8_{\pm2.4}$ | $80.2_{\pm1.4}$ | $63.7_{\pm1.1}/48.0_{\pm1.0}$ | $90.3_{\pm1.7}$ |
| | SGCN | $66.1_{\pm2.6}/67.5_{\pm2.8}$ | $92.1_{\pm1.6}$ | $65.4_{\pm4.5}/68.5_{\pm3.6}$ | $85.1_{\pm0.9}$ | n.a./n.a. | n.a. | $67.2_{\pm4.2}/61.6_{\pm7.3}$ | $80.1_{\pm0.9}$ | $30.1_{\pm1.1}/27.4_{\pm0.9}$ | $80.1_{\pm0.9}$ | $68.0_{\pm1.2}/75.6_{\pm2.2}$ | $79.3_{\pm2.8}$ | $55.9_{\pm0.9}/53.8_{\pm1.4}$ | $90.7_{\pm1.8}$ |
| | GCN-EBM | $66.1_{\pm3.0}/66.1_{\pm3.0}$ | $91.7_{\pm1.5}$ | $63.1_{\pm4.7}/62.8_{\pm4.5}$ | $83.7_{\pm1.5}$ | n.a./n.a. | n.a. | $65.7_{\pm6.8}/58.5_{\pm7.5}$ | $78.1_{\pm1.0}$ | $29.6_{\pm0.8}/26.4_{\pm0.9}$ | $78.1_{\pm1.0}$ | $67.9_{\pm1.1}/71.8_{\pm2.7}$ | $78.0_{\pm2.6}$ | $55.7_{\pm0.8}/53.4_{\pm0.9}$ | $88.8_{\pm2.7}$ |
| | GCN-HEAT | $66.1_{\pm3.0}/64.0_{\pm3.5}$ | $91.7_{\pm1.5}$ | $63.1_{\pm4.7}/60.4_{\pm6.3}$ | $83.7_{\pm1.5}$ | n.a./n.a. | n.a. | $65.7_{\pm6.8}/59.8_{\pm9.4}$ | $78.1_{\pm1.0}$ | $29.6_{\pm0.8}/78.0_{\pm4.9}$ | $78.1_{\pm1.0}$ | $67.9_{\pm1.1}/60.0_{\pm6.9}$ | $78.0_{\pm2.6}$ | $55.7_{\pm0.8}/59.5_{\pm2.9}$ | $88.8_{\pm2.7}$ |
| | GCNSafe | $66.1_{\pm3.0}/67.4_{\pm3.3}$ | $91.7_{\pm1.5}$ | $63.1_{\pm4.7}/67.4_{\pm4.7}$ | $83.7_{\pm1.5}$ | n.a./n.a. | n.a. | $65.7_{\pm6.8}/53.8_{\pm2.5}$ | $78.1_{\pm1.0}$ | $29.6_{\pm0.8}/38.4_{\pm1.2}$ | $78.1_{\pm1.0}$ | $67.9_{\pm1.1}/49.4_{\pm0.8}$ | $78.0_{\pm2.6}$ | $55.7_{\pm0.8}/60.3_{\pm1.3}$ | $88.8_{\pm2.7}$ |
| | **GCN-GEBM** | $66.1_{\pm3.0}/68.4_{\pm3.0}$ | $91.7_{\pm1.5}$ | $63.1_{\pm4.7}/66.3_{\pm3.7}$ | $83.7_{\pm1.5}$ | n.a./n.a. | n.a. | $65.7_{\pm6.8}/84.0_{\pm5.1}$ | $78.1_{\pm1.0}$ | $29.6_{\pm0.8}/73.0_{\pm6.2}$ | $78.1_{\pm1.0}$ | $67.9_{\pm1.1}/49.6_{\pm0.8}$ | $78.0_{\pm2.6}$ | $55.7_{\pm0.8}/57.1_{\pm1.9}$ | $88.8_{\pm2.7}$ |
| Amazon Computers | GCN-MCD | $76.1_{\pm6.0}/49.5_{\pm1.4}$ | $88.0_{\pm1.5}$ | $86.6_{\pm2.5}/50.0_{\pm1.3}$ | $93.5_{\pm2.4}$ | $57.9_{\pm1.5}/50.1_{\pm1.1}$ | $81.7_{\pm1.1}$ | $54.7_{\pm2.4}/50.2_{\pm1.2}$ | $81.7_{\pm1.1}$ | $42.2_{\pm1.5}/49.6_{\pm1.0}$ | $81.7_{\pm1.1}$ | $82.7_{\pm0.9}/49.8_{\pm1.3}$ | $82.7_{\pm1.1}$ | $62.2_{\pm0.6}/50.1_{\pm0.9}$ | $92.3_{\pm0.7}$ |
| | GCN-DE | $76.2_{\pm5.8}/75.6_{\pm3.4}$ | $87.2_{\pm1.2}$ | $86.7_{\pm2.1}/79.8_{\pm1.6}$ | $93.0_{\pm2.7}$ | $58.6_{\pm1.5}/52.9_{\pm1.5}$ | $80.7_{\pm1.2}$ | $55.2_{\pm2.0}/52.4_{\pm1.6}$ | $80.7_{\pm1.2}$ | $41.4_{\pm1.4}/52.9_{\pm1.4}$ | $80.7_{\pm1.2}$ | $80.3_{\pm0.7}/78.4_{\pm0.5}$ | $82.0_{\pm1.0}$ | $62.5_{\pm0.6}/55.2_{\pm0.8}$ | $90.9_{\pm0.8}$ |
| | GCN-MCD+DE | $76.8_{\pm4.9}/75.5_{\pm3.0}$ | $87.1_{\pm1.2}$ | $87.2_{\pm2.5}/80.2_{\pm1.4}$ | $92.8_{\pm2.7}$ | $58.5_{\pm1.3}/53.0_{\pm1.5}$ | $80.7_{\pm1.3}$ | $55.4_{\pm2.3}/52.2_{\pm1.7}$ | $80.7_{\pm1.3}$ | $41.8_{\pm1.6}/52.7_{\pm1.6}$ | $80.7_{\pm1.3}$ | $80.4_{\pm0.6}/78.3_{\pm0.6}$ | $81.7_{\pm1.5}$ | $62.6_{\pm0.8}/55.3_{\pm0.7}$ | $91.1_{\pm0.7}$ |
| | GCN-BNN | $69.4_{\pm9.8}/73.8_{\pm10.0}$ | $85.4_{\pm3.2}$ | $83.9_{\pm5.7}/85.4_{\pm5.5}$ | $92.3_{\pm3.4}$ | $57.1_{\pm1.0}/55.7_{\pm1.5}$ | $77.6_{\pm2.6}$ | $53.2_{\pm2.2}/53.1_{\pm2.4}$ | $77.6_{\pm2.6}$ | $41.9_{\pm1.5}/57.0_{\pm1.8}$ | $77.6_{\pm2.6}$ | $78.5_{\pm1.2}/50.5_{\pm5.2}$ | $80.7_{\pm2.4}$ | $61.8_{\pm1.3}/64.8_{\pm1.6}$ | $90.9_{\pm1.6}$ |
| | GCN-Ens | $76.3_{\pm6.4}/73.3_{\pm6.0}$ | $88.1_{\pm1.7}$ | $86.5_{\pm2.8}/84.1_{\pm4.3}$ | $93.5_{\pm2.6}$ | $58.1_{\pm1.4}/48.8_{\pm1.8}$ | $81.7_{\pm0.9}$ | $54.9_{\pm2.3}/51.5_{\pm2.0}$ | $81.7_{\pm0.9}$ | $41.0_{\pm1.4}/56.2_{\pm0.7}$ | $81.7_{\pm0.9}$ | $82.9_{\pm0.7}/65.1_{\pm3.9}$ | $82.6_{\pm1.0}$ | $62.2_{\pm0.5}/61.2_{\pm1.5}$ | $92.4_{\pm0.6}$ |
| | GPN | $86.8_{\pm3.0}/80.8_{\pm3.0}$ | $86.0_{\pm1.4}$ | $86.4_{\pm2.2}/84.4_{\pm1.4}$ | $94.4_{\pm1.1}$ | $56.5_{\pm1.0}/56.6_{\pm1.4}$ | $79.5_{\pm2.4}$ | $56.5_{\pm1.1}/57.1_{\pm1.5}$ | $79.5_{\pm2.4}$ | $55.8_{\pm1.0}/58.0_{\pm1.5}$ | $79.5_{\pm2.4}$ | $51.5_{\pm1.2}/76.5_{\pm1.3}$ | $79.3_{\pm2.0}$ | $67.4_{\pm2.8}/51.7_{\pm1.5}$ | $86.1_{\pm14.4}$ |
| | SGCN | $58.5_{\pm17.3}/67.9_{\pm13.5}$ | $58.0_{\pm29.3}$ | $78.4_{\pm8.5}/84.6_{\pm7.9}$ | $92.7_{\pm6.5}$ | $49.4_{\pm1.2}/50.6_{\pm1.2}$ | $16.1_{\pm15.1}$ | $48.6_{\pm1.3}/51.4_{\pm1.2}$ | $16.1_{\pm15.1}$ | $47.1_{\pm2.9}/45.7_{\pm2.5}$ | $16.1_{\pm15.1}$ | $94.1_{\pm0.9}/13.1_{\pm23.8}$ | $14.9_{\pm13.4}$ | $43.4_{\pm0.4}/56.3_{\pm1.3}$ | $16.7_{\pm10.6}$ |
| | GCN-EBM | $76.0_{\pm6.3}/75.3_{\pm4.3}$ | $88.0_{\pm1.8}$ | $86.3_{\pm2.6}/84.5_{\pm3.2}$ | $93.5_{\pm2.5}$ | $58.0_{\pm1.5}/57.8_{\pm1.6}$ | $81.9_{\pm1.0}$ | $55.0_{\pm2.4}/54.6_{\pm2.3}$ | $81.9_{\pm1.0}$ | $42.3_{\pm1.4}/40.8_{\pm1.4}$ | $81.9_{\pm1.0}$ | $82.9_{\pm0.8}/87.2_{\pm1.5}$ | $82.6_{\pm1.2}$ | $62.3_{\pm0.5}/60.6_{\pm0.8}$ | $92.4_{\pm0.7}$ |
| | GCN-HEAT | $76.0_{\pm6.3}/73.7_{\pm6.0}$ | $88.0_{\pm1.8}$ | $86.3_{\pm2.6}/84.3_{\pm3.0}$ | $93.5_{\pm2.5}$ | $58.0_{\pm1.5}/57.0_{\pm1.8}$ | $81.9_{\pm1.0}$ | $55.0_{\pm2.4}/55.2_{\pm2.9}$ | $81.9_{\pm1.0}$ | $42.3_{\pm1.4}/58.4_{\pm1.3}$ | $81.9_{\pm1.0}$ | $82.9_{\pm0.8}/79.6_{\pm4.8}$ | $82.6_{\pm1.2}$ | $62.3_{\pm0.5}/63.1_{\pm1.4}$ | $92.4_{\pm0.7}$ |
| | GCNSafe | $76.0_{\pm6.3}/87.6_{\pm2.5}$ | $88.0_{\pm1.8}$ | $86.3_{\pm2.6}/84.3_{\pm1.9}$ | $93.5_{\pm2.5}$ | $58.0_{\pm1.5}/53.3_{\pm1.4}$ | $81.9_{\pm1.0}$ | $55.0_{\pm2.4}/52.0_{\pm1.3}$ | $81.9_{\pm1.0}$ | $42.3_{\pm1.4}/43.1_{\pm1.4}$ | $81.9_{\pm1.0}$ | $82.9_{\pm0.8}/59.3_{\pm1.3}$ | $82.6_{\pm1.2}$ | $62.3_{\pm0.5}/55.2_{\pm0.9}$ | $92.4_{\pm0.7}$ |
| | **GCN-GEBM** | $76.0_{\pm6.3}/84.8_{\pm2.8}$ | $88.0_{\pm1.8}$ | $86.3_{\pm2.6}/90.1_{\pm1.5}$ | $93.5_{\pm2.5}$ | $58.0_{\pm1.5}/84.3_{\pm2.8}$ | $81.9_{\pm1.0}$ | $55.0_{\pm2.4}/72.9_{\pm5.3}$ | $81.9_{\pm1.0}$ | $42.3_{\pm1.4}/92.0_{\pm7.7}$ | $81.9_{\pm1.0}$ | $82.9_{\pm0.8}/52.1_{\pm1.1}$ | $82.6_{\pm1.2}$ | $62.3_{\pm0.5}/61.9_{\pm1.5}$ | $92.4_{\pm0.7}$ |
| Amazon Photo | GCN-MCD | $75.5_{\pm4.2}/49.9_{\pm1.4}$ | $92.0_{\pm0.9}$ | $82.4_{\pm4.6}/49.8_{\pm1.5}$ | $96.3_{\pm0.2}$ | $57.8_{\pm1.5}/50.1_{\pm1.4}$ | $90.2_{\pm2.0}$ | $52.1_{\pm2.1}/49.7_{\pm1.3}$ | $90.2_{\pm2.0}$ | $42.9_{\pm1.7}/49.7_{\pm1.1}$ | $90.2_{\pm2.0}$ | $84.6_{\pm0.6}/50.3_{\pm1.5}$ | $91.8_{\pm0.6}$ | $62.0_{\pm1.1}/49.5_{\pm1.5}$ | $97.6_{\pm0.7}$ |
| | GCN-DE | $76.6_{\pm3.8}/61.3_{\pm2.6}$ | $91.3_{\pm1.1}$ | $81.7_{\pm4.5}/74.0_{\pm3.9}$ | $95.9_{\pm0.4}$ | $59.0_{\pm1.5}/54.0_{\pm1.6}$ | $89.5_{\pm1.9}$ | $52.4_{\pm1.7}/52.0_{\pm2.0}$ | $89.5_{\pm1.9}$ | $42.4_{\pm1.2}/51.1_{\pm1.3}$ | $89.5_{\pm1.9}$ | $82.0_{\pm0.6}/82.0_{\pm0.5}$ | $91.2_{\pm0.7}$ | $61.3_{\pm1.3}/54.0_{\pm1.1}$ | $96.9_{\pm0.8}$ |
| | GCN-MCD+DE | $76.1_{\pm4.8}/61.5_{\pm3.1}$ | $91.5_{\pm1.0}$ | $82.0_{\pm4.6}/74.1_{\pm3.8}$ | $95.9_{\pm0.5}$ | $59.0_{\pm1.4}/54.2_{\pm1.6}$ | $89.5_{\pm1.7}$ | $52.7_{\pm2.2}/52.2_{\pm2.0}$ | $89.5_{\pm1.7}$ | $42.2_{\pm1.4}/51.5_{\pm1.5}$ | $89.5_{\pm1.7}$ | $82.0_{\pm0.5}/82.0_{\pm0.6}$ | $91.2_{\pm0.7}$ | $61.3_{\pm1.1}/54.0_{\pm1.1}$ | $96.9_{\pm0.8}$ |
| | GCN-BNN | $78.9_{\pm6.3}/80.8_{\pm7.9}$ | $93.2_{\pm0.8}$ | $86.0_{\pm4.6}/88.6_{\pm4.9}$ | $96.5_{\pm0.4}$ | $56.2_{\pm1.3}/55.5_{\pm1.0}$ | $90.0_{\pm2.6}$ | $51.1_{\pm3.3}/50.1_{\pm3.1}$ | $90.0_{\pm2.6}$ | $43.9_{\pm1.1}/58.0_{\pm1.6}$ | $90.0_{\pm2.6}$ | $77.6_{\pm2.2}/65.3_{\pm4.5}$ | $90.7_{\pm1.4}$ | $65.4_{\pm1.9}/67.1_{\pm2.0}$ | $97.9_{\pm0.9}$ |
| | GCN-Ens | $75.9_{\pm4.4}/76.8_{\pm8.3}$ | $92.0_{\pm0.8}$ | $82.4_{\pm4.5}/82.4_{\pm4.7}$ | $96.3_{\pm0.2}$ | $58.1_{\pm1.5}/53.4_{\pm1.6}$ | $90.4_{\pm2.0}$ | $51.9_{\pm2.0}/50.1_{\pm1.2}$ | $90.4_{\pm2.0}$ | $43.3_{\pm1.5}/56.0_{\pm1.1}$ | $90.4_{\pm2.0}$ | $84.6_{\pm0.7}/75.8_{\pm2.8}$ | $91.9_{\pm0.5}$ | $62.1_{\pm1.1}/59.6_{\pm1.9}$ | $97.7_{\pm0.6}$ |
| | GPN | $78.6_{\pm5.8}/88.6_{\pm2.7}$ | $89.5_{\pm1.2}$ | $91.0_{\pm1.7}/79.3_{\pm4.5}$ | $94.3_{\pm1.6}$ | $54.0_{\pm1.7}/54.6_{\pm1.8}$ | $86.2_{\pm2.6}$ | $54.0_{\pm1.7}/54.6_{\pm1.8}$ | $86.2_{\pm2.6}$ | $53.0_{\pm1.5}/55.2_{\pm2.0}$ | $86.2_{\pm2.6}$ | $51.2_{\pm0.6}/74.1_{\pm2.3}$ | $88.8_{\pm1.9}$ | $68.2_{\pm1.4}/49.9_{\pm2.1}$ | $96.4_{\pm0.9}$ |
| | SGCN | $76.5_{\pm4.3}/80.2_{\pm4.9}$ | $91.9_{\pm0.9}$ | $77.5_{\pm5.0}/78.2_{\pm5.4}$ | $96.0_{\pm1.0}$ | $49.3_{\pm1.5}/50.8_{\pm1.6}$ | $11.4_{\pm7.5}$ | $48.5_{\pm1.5}/51.8_{\pm1.5}$ | $11.4_{\pm7.5}$ | $46.9_{\pm2.5}/46.7_{\pm2.1}$ | $11.4_{\pm7.5}$ | $86.2_{\pm5.0}/63.5_{\pm34.8}$ | $64.8_{\pm36.0}$ | $42.8_{\pm0.9}/56.1_{\pm3.8}$ | $12.2_{\pm7.0}$ |
| | GCN-EBM | $75.8_{\pm4.3}/75.2_{\pm4.7}$ | $92.0_{\pm0.8}$ | $82.3_{\pm4.5}/75.9_{\pm3.7}$ | $96.3_{\pm0.2}$ | $57.9_{\pm1.4}/58.5_{\pm1.5}$ | $90.2_{\pm1.9}$ | $51.9_{\pm2.1}/51.5_{\pm1.4}$ | $90.2_{\pm1.9}$ | $42.6_{\pm1.6}/41.3_{\pm1.6}$ | $90.2_{\pm1.9}$ | $84.6_{\pm0.7}/90.5_{\pm0.9}$ | $91.9_{\pm0.6}$ | $62.1_{\pm1.1}/56.8_{\pm1.4}$ | $97.7_{\pm0.6}$ |
| | GCN-HEAT | $75.8_{\pm4.3}/74.7_{\pm7.9}$ | $92.0_{\pm0.8}$ | $82.3_{\pm4.5}/72.9_{\pm8.4}$ | $96.3_{\pm0.2}$ | $57.9_{\pm1.4}/57.1_{\pm1.6}$ | $90.2_{\pm1.9}$ | $51.9_{\pm2.1}/50.9_{\pm1.8}$ | $90.2_{\pm1.9}$ | $42.6_{\pm1.6}/56.6_{\pm1.8}$ | $90.2_{\pm1.9}$ | $84.6_{\pm0.7}/82.4_{\pm4.7}$ | $91.9_{\pm0.6}$ | $62.1_{\pm1.1}/59.3_{\pm2.3}$ | $97.7_{\pm0.6}$ |
| | GCNSafe | $75.8_{\pm4.3}/75.8_{\pm6.8}$ | $92.0_{\pm0.8}$ | $82.3_{\pm4.5}/83.2_{\pm4.3}$ | $96.3_{\pm0.2}$ | $57.9_{\pm1.4}/55.3_{\pm1.0}$ | $90.2_{\pm1.9}$ | $51.9_{\pm2.1}/51.7_{\pm1.2}$ | $90.2_{\pm1.9}$ | $42.6_{\pm1.6}/44.5_{\pm2.0}$ | $90.2_{\pm1.9}$ | $84.6_{\pm0.7}/63.4_{\pm0.8}$ | $91.9_{\pm0.6}$ | $62.1_{\pm1.1}/55.8_{\pm3.1}$ | $97.7_{\pm0.6}$ |
| | **GCN-GEBM** | $75.8_{\pm4.3}/84.2_{\pm5.3}$ | $92.0_{\pm0.8}$ | $82.3_{\pm4.5}/87.4_{\pm3.9}$ | $96.3_{\pm0.2}$ | $57.9_{\pm1.4}/83.3_{\pm3.9}$ | $90.2_{\pm1.9}$ | $51.9_{\pm2.1}/58.4_{\pm8.0}$ | $90.2_{\pm1.9}$ | $42.6_{\pm1.6}/92.0_{\pm2.7}$ | $90.2_{\pm1.9}$ | $84.6_{\pm0.7}/57.8_{\pm1.2}$ | $91.9_{\pm0.6}$ | $62.1_{\pm1.1}/62.2_{\pm1.9}$ | $97.7_{\pm0.6}$ |
| Coauthor-CS | GCN-MCD | $88.7_{\pm1.0}/50.1_{\pm1.4}$ | $93.1_{\pm0.3}$ | $90.9_{\pm2.3}/49.6_{\pm1.0}$ | $94.8_{\pm0.4}$ | $68.4_{\pm0.5}/50.2_{\pm0.7}$ | $92.1_{\pm0.3}$ | $69.7_{\pm0.7}/50.1_{\pm0.7}$ | $92.1_{\pm0.3}$ | $36.5_{\pm1.2}/50.1_{\pm1.0}$ | $92.1_{\pm0.3}$ | $76.6_{\pm0.7}/50.2_{\pm1.0}$ | $92.6_{\pm0.5}$ | $64.9_{\pm0.4}/49.7_{\pm0.9}$ | $98.8_{\pm0.4}$ |
| | GCN-DE | $87.8_{\pm1.4}/75.6_{\pm1.4}$ | $92.2_{\pm0.5}$ | $90.1_{\pm2.1}/81.1_{\pm2.0}$ | $94.0_{\pm0.5}$ | $68.8_{\pm0.7}/61.6_{\pm0.8}$ | $90.9_{\pm0.6}$ | $65.9_{\pm1.3}/68.6_{\pm0.9}$ | $90.9_{\pm0.6}$ | $35.2_{\pm1.0}/58.6_{\pm0.8}$ | $90.9_{\pm0.6}$ | $70.9_{\pm0.8}/76.6_{\pm0.7}$ | $90.9_{\pm0.6}$ | $63.4_{\pm0.5}/58.1_{\pm0.7}$ | $98.2_{\pm0.3}$ |
| | GCN-MCD+DE | $88.0_{\pm1.6}/75.4_{\pm1.8}$ | $92.2_{\pm0.4}$ | $89.7_{\pm2.5}/80.8_{\pm2.1}$ | $93.8_{\pm0.6}$ | $68.5_{\pm0.8}/61.4_{\pm1.0}$ | $90.8_{\pm0.6}$ | $66.2_{\pm1.4}/68.6_{\pm0.9}$ | $90.8_{\pm0.6}$ | $35.2_{\pm1.0}/58.8_{\pm1.0}$ | $90.8_{\pm0.6}$ | $70.9_{\pm0.7}/76.7_{\pm0.6}$ | $90.9_{\pm0.7}$ | $63.4_{\pm0.5}/58.1_{\pm0.5}$ | $98.2_{\pm0.3}$ |
| | GCN-BNN | $85.1_{\pm3.6}/84.4_{\pm3.8}$ | $91.9_{\pm0.5}$ | $87.9_{\pm3.5}/86.6_{\pm3.9}$ | $94.2_{\pm0.8}$ | $66.8_{\pm0.8}/67.1_{\pm0.8}$ | $91.2_{\pm0.5}$ | $55.7_{\pm5.2}/71.2_{\pm4.0}$ | $91.2_{\pm0.5}$ | $29.3_{\pm1.1}/53.9_{\pm1.1}$ | $91.2_{\pm0.5}$ | $69.1_{\pm1.0}/61.1_{\pm1.3}$ | $91.7_{\pm0.8}$ | $70.3_{\pm1.1}/72.0_{\pm1.4}$ | $98.8_{\pm0.5}$ |
| | GCN-Ens | $88.8_{\pm1.1}/86.5_{\pm1.9}$ | $93.1_{\pm0.3}$ | $91.0_{\pm2.2}/89.3_{\pm2.4}$ | $94.8_{\pm0.4}$ | $68.5_{\pm0.4}/63.9_{\pm1.8}$ | $92.2_{\pm0.2}$ | $69.9_{\pm0.6}/70.9_{\pm1.1}$ | $92.2_{\pm0.2}$ | $34.0_{\pm0.7}/60.5_{\pm1.2}$ | $92.2_{\pm0.2}$ | $76.7_{\pm0.6}/76.2_{\pm1.2}$ | $92.7_{\pm0.5}$ | $65.0_{\pm0.4}/66.9_{\pm0.7}$ | $98.9_{\pm0.4}$ |
| | GPN | $73.3_{\pm3.5}/89.8_{\pm2.0}$ | $86.7_{\pm0.9}$ | $83.9_{\pm3.9}/81.1_{\pm4.9}$ | $91.7_{\pm1.3}$ | $56.5_{\pm1.0}/58.0_{\pm1.5}$ | $85.4_{\pm1.7}$ | $54.7_{\pm0.8}/58.9_{\pm1.8}$ | $85.4_{\pm1.7}$ | $54.6_{\pm0.8}/58.9_{\pm1.8}$ | $85.4_{\pm1.7}$ | $48.7_{\pm0.3}/76.3_{\pm1.6}$ | $86.7_{\pm1.5}$ | $70.4_{\pm1.5}/48.4_{\pm1.1}$ | $94.8_{\pm1.2}$ |
| | SGCN | $49.1_{\pm9.9}/62.3_{\pm8.5}$ | $29.0_{\pm18.1}$ | $61.0_{\pm9.4}/41.5_{\pm8.7}$ | $30.3_{\pm13.5}$ | $47.8_{\pm1.0}/52.9_{\pm0.8}$ | $14.0_{\pm8.5}$ | $33.9_{\pm1.6}/66.5_{\pm1.1}$ | $14.0_{\pm8.5}$ | $34.2_{\pm1.9}/46.4_{\pm2.5}$ | $14.0_{\pm8.5}$ | $84.3_{\pm4.4}/14.9_{\pm2.1}$ | $15.1_{\pm9.2}$ | $38.0_{\pm1.8}/64.0_{\pm1.0}$ | $13.1_{\pm9.3}$ |
| | GCN-EBM | $88.7_{\pm1.1}/88.6_{\pm1.1}$ | $93.1_{\pm0.4}$ | $90.9_{\pm2.1}/91.8_{\pm1.9}$ | $94.8_{\pm0.4}$ | $68.4_{\pm0.5}/68.7_{\pm0.9}$ | $92.0_{\pm0.2}$ | $69.5_{\pm0.8}/70.9_{\pm1.3}$ | $92.0_{\pm0.2}$ | $36.3_{\pm0.8}/29.4_{\pm0.8}$ | $92.0_{\pm0.2}$ | $76.6_{\pm0.7}/81.5_{\pm1.2}$ | $92.6_{\pm0.5}$ | $64.9_{\pm0.4}/59.5_{\pm0.3}$ | $98.9_{\pm0.4}$ |
| | GCN-HEAT | $88.7_{\pm1.1}/88.1_{\pm1.9}$ | $93.1_{\pm0.4}$ | $90.9_{\pm2.1}/88.2_{\pm5.3}$ | $94.8_{\pm0.4}$ | $68.4_{\pm0.5}/68.3_{\pm1.3}$ | $92.0_{\pm0.2}$ | $69.5_{\pm0.8}/70.8_{\pm1.1}$ | $92.0_{\pm0.2}$ | $36.3_{\pm0.8}/69.4_{\pm1.0}$ | $92.0_{\pm0.2}$ | $76.6_{\pm0.7}/71.2_{\pm4.3}$ | $92.6_{\pm0.5}$ | $64.9_{\pm0.4}/66.5_{\pm1.3}$ | $98.9_{\pm0.4}$ |
| | GCNSafe | $88.7_{\pm1.1}/92.0_{\pm1.4}$ | $93.1_{\pm0.4}$ | $90.9_{\pm2.1}/95.6_{\pm1.1}$ | $94.8_{\pm0.4}$ | $68.4_{\pm0.5}/61.3_{\pm0.9}$ | $92.0_{\pm0.2}$ | $69.5_{\pm0.8}/63.1_{\pm1.0}$ | $92.0_{\pm0.2}$ | $36.3_{\pm0.8}/37.5_{\pm0.8}$ | $92.0_{\pm0.2}$ | $76.6_{\pm0.7}/61.6_{\pm0.7}$ | $92.6_{\pm0.5}$ | $64.9_{\pm0.4}/62.0_{\pm0.4}$ | $98.9_{\pm0.4}$ |
| | **GCN-GEBM** | $88.7_{\pm1.1}/92.7_{\pm1.3}$ | $93.1_{\pm0.4}$ | $90.9_{\pm2.1}/93.6_{\pm2.9}$ | $94.8_{\pm0.4}$ | $68.4_{\pm0.5}/96.0_{\pm0.6}$ | $92.0_{\pm0.2}$ | $69.5_{\pm0.8}/99.2_{\pm0.5}$ | $92.0_{\pm0.2}$ | $36.3_{\pm0.8}/95.1_{\pm2.0}$ | $92.0_{\pm0.2}$ | $76.6_{\pm0.7}/56.0_{\pm0.7}$ | $92.6_{\pm0.5}$ | $64.9_{\pm0.4}/68.4_{\pm1.1}$ | $98.9_{\pm0.4}$ |
| Coauthor-Physics | GCN-MCD | $91.4_{\pm1.2}/50.3_{\pm0.8}$ | $96.6_{\pm1.2}$ | $94.6_{\pm1.6}/49.9_{\pm0.8}$ | $98.1_{\pm0.3}$ | $59.7_{\pm1.1}/50.0_{\pm0.5}$ | $92.9_{\pm0.7}$ | $62.3_{\pm0.8}/50.1_{\pm0.7}$ | $92.9_{\pm0.7}$ | $43.2_{\pm0.5}/50.0_{\pm0.6}$ | $92.9_{\pm0.7}$ | $80.1_{\pm0.7}/49.9_{\pm0.7}$ | $94.5_{\pm0.8}$ | $58.6_{\pm0.4}/50.1_{\pm0.7}$ | $98.8_{\pm0.2}$ |
| | GCN-DE | $91.2_{\pm1.0}/85.2_{\pm1.2}$ | $96.5_{\pm1.0}$ | $93.9_{\pm1.6}/89.7_{\pm1.4}$ | $97.9_{\pm0.4}$ | $60.1_{\pm1.0}/56.5_{\pm0.8}$ | $92.3_{\pm0.6}$ | $63.1_{\pm0.7}/60.2_{\pm0.8}$ | $92.3_{\pm0.6}$ | $42.9_{\pm1.0}/50.9_{\pm0.7}$ | $92.3_{\pm0.6}$ | $75.6_{\pm0.7}/78.8_{\pm0.7}$ | $94.1_{\pm0.7}$ | $58.4_{\pm0.4}/56.1_{\pm0.4}$ | $98.4_{\pm0.3}$ |
| | GCN-MCD+DE | $90.8_{\pm1.0}/84.7_{\pm1.1}$ | $96.5_{\pm1.0}$ | $94.1_{\pm1.6}/90.0_{\pm1.3}$ | $97.9_{\pm0.3}$ | $60.0_{\pm1.1}/56.3_{\pm0.9}$ | $92.3_{\pm0.7}$ | $62.8_{\pm1.0}/60.3_{\pm0.9}$ | $92.3_{\pm0.7}$ | $42.2_{\pm0.7}/50.4_{\pm0.6}$ | $92.3_{\pm0.7}$ | $75.6_{\pm0.6}/78.8_{\pm0.7}$ | $94.2_{\pm0.7}$ | $58.3_{\pm0.4}/56.1_{\pm0.3}$ | $98.4_{\pm0.3}$ |
| | GCN-BNN | $89.3_{\pm2.4}/88.6_{\pm2.3}$ | $96.6_{\pm1.0}$ | $91.1_{\pm3.5}/91.7_{\pm3.5}$ | $97.9_{\pm0.3}$ | $59.6_{\pm1.5}/59.7_{\pm1.8}$ | $92.4_{\pm0.9}$ | $51.8_{\pm4.7}/62.4_{\pm5.2}$ | $92.4_{\pm0.9}$ | $35.8_{\pm0.9}/54.1_{\pm1.5}$ | $92.4_{\pm0.9}$ | $74.7_{\pm1.8}/67.1_{\pm3.3}$ | $94.2_{\pm1.0}$ | $62.4_{\pm0.9}/63.6_{\pm1.2}$ | $98.9_{\pm0.3}$ |
| | GCN-Ens | $91.4_{\pm1.2}/86.2_{\pm4.9}$ | $96.7_{\pm1.2}$ | $94.6_{\pm1.6}/90.7_{\pm2.9}$ | $98.1_{\pm0.3}$ | $59.7_{\pm1.0}/56.7_{\pm1.5}$ | $92.9_{\pm0.7}$ | $62.5_{\pm0.6}/60.5_{\pm1.7}$ | $92.9_{\pm0.7}$ | $42.1_{\pm0.5}/53.1_{\pm0.8}$ | $92.9_{\pm0.7}$ | $80.1_{\pm0.7}/78.9_{\pm2.0}$ | $94.5_{\pm0.8}$ | $58.6_{\pm0.4}/61.8_{\pm0.8}$ | $98.7_{\pm0.3}$ |
| | GPN | $85.6_{\pm3.2}/90.2_{\pm2.6}$ | $96.8_{\pm0.5}$ | $83.4_{\pm3.7}/94.7_{\pm1.4}$ | $97.7_{\pm0.3}$ | $58.8_{\pm1.3}/58.8_{\pm1.5}$ | $92.3_{\pm0.7}$ | $56.2_{\pm1.2}/60.1_{\pm1.9}$ | $92.3_{\pm0.7}$ | $55.8_{\pm0.9}/60.2_{\pm1.9}$ | $92.3_{\pm0.7}$ | $52.4_{\pm0.8}/78.0_{\pm1.6}$ | $93.3_{\pm0.8}$ | $69.8_{\pm1.2}/51.5_{\pm0.8}$ | $98.1_{\pm0.4}$ |
| | SGCN | $91.2_{\pm1.3}/90.3_{\pm3.5}$ | $96.5_{\pm1.0}$ | $94.9_{\pm1.2}/95.2_{\pm1.5}$ | $98.1_{\pm0.4}$ | $60.8_{\pm1.1}/62.1_{\pm1.6}$ | $92.4_{\pm0.9}$ | $62.3_{\pm1.0}/62.2_{\pm2.7}$ | $92.4_{\pm0.9}$ | $42.2_{\pm0.6}/40.4_{\pm0.7}$ | $92.4_{\pm0.9}$ | $79.2_{\pm0.8}/80.9_{\pm2.1}$ | $94.0_{\pm0.7}$ | $58.7_{\pm0.6}/57.7_{\pm1.4}$ | $98.6_{\pm0.4}$ |
| | GCN-EBM | $91.5_{\pm1.1}/91.5_{\pm1.7}$ | $96.7_{\pm1.1}$ | $94.6_{\pm1.7}/95.3_{\pm1.6}$ | $98.1_{\pm0.3}$ | $59.8_{\pm1.0}/59.7_{\pm1.2}$ | $92.8_{\pm0.7}$ | $62.4_{\pm0.8}/62.0_{\pm1.4}$ | $92.8_{\pm0.7}$ | $43.0_{\pm0.8}/39.6_{\pm0.8}$ | $92.8_{\pm0.7}$ | $80.0_{\pm0.8}/83.0_{\pm1.0}$ | $94.5_{\pm0.7}$ | $58.6_{\pm0.4}/56.5_{\pm0.7}$ | $98.7_{\pm0.2}$ |
| | GCN-HEAT | $91.5_{\pm1.1}/88.9_{\pm2.5}$ | $96.7_{\pm1.1}$ | $94.6_{\pm1.7}/92.7_{\pm1.9}$ | $98.1_{\pm0.3}$ | $59.8_{\pm1.0}/59.2_{\pm1.5}$ | $92.8_{\pm0.7}$ | $62.4_{\pm0.8}/62.5_{\pm1.4}$ | $92.8_{\pm0.7}$ | $43.0_{\pm0.8}/60.1_{\pm0.7}$ | $92.8_{\pm0.7}$ | $80.0_{\pm0.8}/77.2_{\pm2.2}$ | $94.5_{\pm0.7}$ | $58.6_{\pm0.4}/59.4_{\pm0.9}$ | $98.7_{\pm0.2}$ |
| | GCNSafe | $91.5_{\pm1.1}/94.7_{\pm1.3}$ | $96.7_{\pm1.1}$ | $94.6_{\pm1.7}/97.3_{\pm1.0}$ | $98.1_{\pm0.3}$ | $59.8_{\pm1.0}/55.5_{\pm0.9}$ | $92.8_{\pm0.7}$ | $62.4_{\pm0.8}/57.4_{\pm0.8}$ | $92.8_{\pm0.7}$ | $43.0_{\pm0.8}/44.0_{\pm0.8}$ | $92.8_{\pm0.7}$ | $80.0_{\pm0.8}/66.4_{\pm0.6}$ | $94.5_{\pm0.7}$ | $58.6_{\pm0.4}/58.4_{\pm0.8}$ | $98.7_{\pm0.2}$ |
| | **GCN-GEBM** | $91.5_{\pm1.1}/93.7_{\pm1.9}$ | $96.7_{\pm1.1}$ | $94.6_{\pm1.7}/97.2_{\pm1.2}$ | $98.1_{\pm0.3}$ | $59.8_{\pm1.0}/83.8_{\pm2.7}$ | $92.8_{\pm0.7}$ | $62.4_{\pm0.8}/93.4_{\pm2.7}$ | $92.8_{\pm0.7}$ | $43.0_{\pm0.8}/78.6_{\pm3.7}$ | $92.8_{\pm0.7}$ | $80.0_{\pm0.8}/60.9_{\pm0.6}$ | $94.5_{\pm0.7}$ | $58.6_{\pm0.4}/61.9_{\pm1.0}$ | $98.7_{\pm0.2}$ |

Table 8: Out-of-Distribution detection AUC-ROC (↑) using aleatoric or epistemic uncertainty (best and runner-up) in a transductive setting.

**Random-Walk Diffusion.** Random-walk diffusion normalizes the adjacency matrix as $\hat{\boldsymbol{A}} = \boldsymbol{D}^{-1}\boldsymbol{A}$ which can be seen as a stochastic transition matrix of random walk on the graph. A different interpretation is that each node distributes its information uniformly to its neighbours which in turn sum incoming signals. It, too, has a dominant eigenvector that correlates with the node degree: $\boldsymbol{v}_{\max} \propto \boldsymbol{d}$ [12] and also favors high degree nodes when applied to a confidence measure.

**Label-Propagation Diffusion.** Diffusion used by Label Propagation [30] normalizes the adjacency matrix as $\hat{\boldsymbol{A}} = \boldsymbol{A}\boldsymbol{D}^{-1}$ and can be seen as a smoothing procedure: Each node will update its features based on the average of its neighbours. Obviously, it has a constant dominant eigenvector and therefore does not favor nodes based on structural information. We use a repeated convex combination of the identity and this diffusion process at the backbone of our GEBM framework: $P_{\boldsymbol{A}} = (\alpha \boldsymbol{I} + (1-\alpha)\boldsymbol{A}\boldsymbol{D}^{-1})^t$. The smoothing behaviour enforces a uniform distribution of confidence within clusters and therefore is suitable candidate for cluster-level anomaly detection.

### C.7 Synthetic Experiments

We motivate the three energy terms that we use to compose the energy measure of GEBM through synthetic experiments. All of them arise naturally from interleaving graph diffusion and energy

| Model | CoraML | CoraML LLM | Citeseer | PubMed | Amazon Computers | Amazon Photo | Coauthor CS | Coauthor Physics |
|---|---|---|---|---|---|---|---|---|
| GCN-MCD | 10.5/20.3 | 10.2/20.0 | 10.2/19.8 | 8.9/18.5 | 10.4/19.9 | 10.7/20.2 | 10.2/19.3 | 10.7/20.6 |
| GCN-DE | 6.7/14.3 | 8.0/16.9 | 7.5/15.6 | 7.2/13.5 | 6.6/15.4 | 7.3/15.8 | 6.6/14.3 | 7.8/16.6 |
| GCN-MCD+DE | 6.8/14.5 | 7.6/16.5 | 7.2/15.2 | 7.0/13.3 | 6.5/15.3 | 6.6/14.7 | 6.8/13.7 | 7.9/16.7 |
| GCN-BNN | 5.7/11.9 | 6.1/13.7 | 5.3/11.3 | 5.1/11.4 | 4.7/11.9 | **3.8**/**7.9** | 5.1/10.8 | 4.9/11.6 |
| GCN-Ens | 7.4/14.1 | 5.7/14.1 | 5.8/12.4 | 7.7/14.2 | 7.3/15.9 | 5.3/11.5 | 4.3/9.5 | 5.7/12.8 |
| GPN | 6.5/12.8 | 6.1/10.9 | 7.5/15.7 | 5.8/11.3 | 4.8/10.8 | 5.1/11.1 | 6.7/13.1 | 6.2/13.0 |
| SGCN | **3.9**/**6.3** | **4.4**/**7.5** | 4.3/**7.9** | **3.9**/**8.2** | 8.1/17.2 | 6.7/14.0 | 9.2/17.7 | **4.4**/9.9 |
| GCN-EBM | 5.2/9.7 | 5.1/10.7 | 5.1/11.4 | 5.9/13.4 | 4.6/12.5 | 5.6/13.3 | **4.3**/**8.3** | 4.6/**9.4** |
| GCN-HEAT | 4.7/9.3 | 4.8/12.2 | **3.8**/9.2 | 5.1/12.2 | **3.9**/**9.9** | 5.6/14.1 | 4.4/9.2 | 4.9/11.1 |
| GCNSafe | 5.8/9.6 | **4.4**/9.4 | 5.4/9.9 | 5.6/9.1 | 6.5/13.8 | 6.5/13.4 | 5.7/11.0 | 5.9/11.6 |
| **GCN-GEBM** | **2.8 / 4.7** | **3.4 / 5.9** | **3.9 / 6.9** | **3.8 / 5.8** | **2.7 / 6.0** | **3.0 / 5.2** | **2.7 / 4.7** | **3.0 / 5.0** |

Table 9: Average o.o.d. detection rank (AUC-ROC) (↓) of epistemic uncertainty versus other epistemic measures / all uncertainty measures over all distribution shifts in a transductive setting (best and runner-up).

marginalization, as discussed in Section 4.3. We want to highlight that GEBM allows to incorporate additional energy functions as well, e.g. if such information is available in designated downstream applications. As shown in this study, we find that the proposed energy functions already suffice to enable GEBM to be highly sensitive to various distribution shifts.

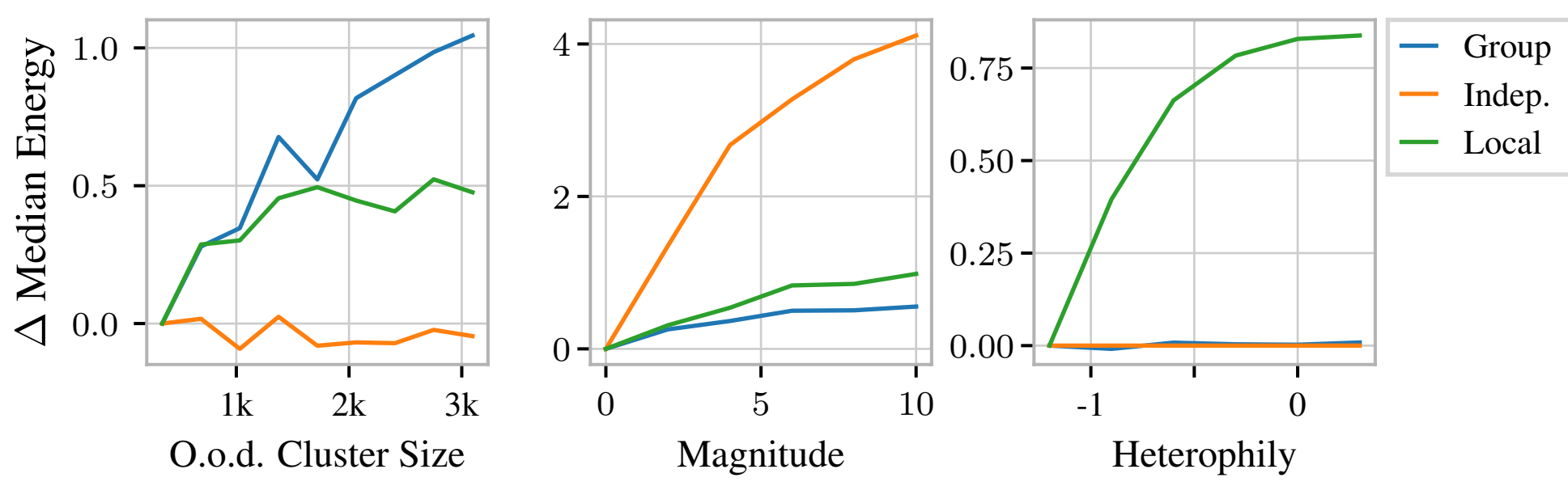

(a) Cluster-Level (b) Node-Level (c) Structure

Figure 4: Energy of different types for anomalies of increasing severity on synthetic data. We vary the size of an o.o.d. cluster on real data (left), insert per-node anomalies to the energies of an SBM (middle), and increase the heterophily in an SBM (right).

**Node-Level Anomalies.** We study node level anomalies by generating synthetic CSBM graphs [57] (intra-edge probability $0.05$, inter-edge probability $0.001$, $n = 1000$, $c = 7$). We then generate $c$-dimensional logits (intuitively confidence scores) according to a standard normal distribution that is centered at $l_{\text{disp}} = 4$ for the dimension of the true class label of a node and $0$ otherwise. This roughly resembles the logits of a well-trained classifier which peak at the true unknown class. We then perturb a fraction $p = 0.05$ of the logits by replacing them with positive noise from a standard normal distribution that is rescaled by an increasing magnitude. For each magnitude, we compute all three energy terms for each node and compare the difference in energy to the corresponding unperturbed energy. Effectively, this monitors how the energy of a node changes under increasingly severe distribution shifts that affect the logits of a classifier. We visualize the median of the energy differences in Figure 4b: While all energies increase with higher magnitudes, the structure-agnostic energy term of GEBM is the most sensitive to anomalies that are located at individual nodes. It does not suffer from the smoothing of anomalous scores induced by graph diffusion.

**Cluster-Level Anomalies.** We study anomalies that affect an entire cluster of nodes by iteratively introducing an anomalous cluster to the graph. To that end, we leave out $k$ classes and train a GCN on the Amazon Photos dataset. At inference, we re-introduce nodes (and edges) of the left-out classes which can be seen as anomalous from the perspective of the classifier. Similar to the aforementioned node-level anomaly, we monitor how each energy type changes while iteratively introducing the entire cluster(s) of o.o.d. nodes into the graph. Again, we see that the cluster-level energy term of GEBM is the most sensitive to this distribution shift in Figure 4a.

| Dataset | Model | LoC (last) AUC-PR↑ (Alea. / Epi.) | Acc.↑ | LoC (hetero.) AUC-PR↑ (Alea. / Epi.) | Acc.↑ | Ber($\hat{p}$) (near) AUC-PR↑ (Alea. / Epi.) | Acc.↑ | Ber(0.5) AUC-PR↑ (Alea. / Epi.) | Acc.↑ | $\mathcal{N}(0,1)$ (far) AUC-PR↑ (Alea. / Epi.) | Acc.↑ | Page Rank AUC-PR↑ (Alea. / Epi.) | Acc.↑ | Homophily AUC-PR↑ (Alea. / Epi.) | Acc.↑ |
|---|---|---|---|---|---|---|---|---|---|---|---|---|---|---|---|
| CoraML | GCN-MCD | 85.1±0.8/46.5±2.3 | 90.4±2.0 | 74.0±4.3/33.4±2.3 | 87.4±1.3 | 64.6±2.6/51.4±3.3 | 75.6±2.3 | 54.2±3.1/51.0±2.7 | 75.6±2.3 | 37.8±1.7/47.6±2.5 | 75.6±2.3 | 71.8±2.2/56.7±2.4 | 82.3±1.5 | 66.1±1.4/50.6±2.4 | 91.4±1.8 |
| | GCN-DE | 81.0±1.8/62.9±2.6 | 87.2±1.7 | 66.8±5.2/41.2±2.6 | 82.0±2.3 | 61.7±3.3/54.1±2.5 | 71.9±2.7 | 45.5±3.9/69.9±2.3 | 71.9±2.7 | 36.0±1.9/61.5±2.6 | 71.9±2.7 | 66.3±2.1/68.4±1.6 | 79.3±2.1 | 66.9±2.2/48.8±1.4 | 85.6±1.9 |
| | GCN-MCD+DE | 81.3±1.9/62.3±2.1 | 87.3±1.6 | 69.0±3.7/40.5±2.4 | 83.0±2.2 | 61.8±3.3/53.7±1.9 | 72.1±3.1 | 45.5±4.0/70.4±2.4 | 72.1±3.1 | 36.4±2.2/62.2±2.7 | 72.1±3.1 | 66.6±2.2/68.4±1.9 | 78.9±2.2 | 67.1±1.4/48.6±1.2 | 85.9±2.1 |
| | GCN-BNN | 81.0±3.0/77.8±3.1 | 89.0±1.4 | 65.0±7.1/62.4±5.9 | 84.9±3.0 | 63.0±3.5/56.9±2.5 | 73.1±2.3 | 45.9±3.3/65.7±4.2 | 73.1±2.3 | 38.1±1.7/58.0±2.6 | 73.1±2.3 | 69.0±2.0/54.3±1.8 | 78.9±1.8 | 65.6±1.4/68.3±1.9 | 88.6±1.9 |
| | GCN-Ens | 85.1±0.8/67.9±3.0 | 90.4±2.0 | 73.8±4.7/49.1±6.4 | 87.3±1.0 | 65.6±2.0/51.5±2.2 | 75.9±2.3 | 54.7±2.0/60.8±2.3 | 75.9±2.3 | 37.4±1.9/54.7±3.1 | 75.9±2.3 | 71.8±2.1/54.3±2.3 | 82.7±1.3 | 66.2±1.1/65.8±2.0 | 91.5±1.8 |
| | GPN | 77.1±4.4/82.2±1.7 | 88.5±1.4 | 71.9±8.7/62.8±5.4 | 86.4±2.7 | 53.0±2.2/53.2±1.4 | 72.8±3.0 | 51.4±2.8/51.2±2.9 | 72.8±3.0 | 52.4±1.5/57.1±3.6 | 72.8±3.0 | 55.3±2.1/62.0±2.7 | 78.4±3.2 | 60.2±4.0/45.7±1.7 | 87.6±2.0 |
| | SGCN | 84.1±2.3/83.2±3.5 | 90.2±1.9 | 78.3±4.1/75.3±10.4 | 89.9±1.4 | 52.7±4.2/51.4±4.2 | 36.2±24.1 | 42.2±7.5/63.7±2.5 | 36.2±24.1 | 37.1±2.1/51.5±9.4 | 36.2±24.1 | 73.3±5.3/57.0±9.7 | 74.8±22.3 | 66.3±1.6/70.6±3.0 | 90.0±1.4 |
| | GCN-EBM | 85.0±0.9/85.5±1.8 | 90.3±1.9 | 73.7±5.0/75.1±5.5 | 87.4±1.3 | 66.0±3.9/63.8±3.7 | 75.8±2.2 | 55.1±2.3/53.9±3.9 | 75.8±2.2 | 37.9±1.8/35.9±1.8 | 75.8±2.2 | 71.9±2.2/71.6±3.5 | 82.6±1.5 | 66.0±1.5/66.3±2.4 | 91.2±1.6 |
| | GCN-HEAT | 85.0±0.9/79.5±5.0 | 90.3±1.9 | 73.7±5.0/70.4±7.6 | 87.4±1.3 | 66.0±3.9/61.2±1.8 | 75.8±2.2 | 55.1±2.3/59.1±3.7 | 75.8±2.2 | 37.9±1.8/73.7±1.6 | 75.8±2.2 | 71.9±2.2/67.3±3.0 | 82.6±1.5 | 66.0±1.5/63.8±2.1 | 91.2±1.6 |
| | GCNSafe | 85.0±0.9/87.3±2.2 | 90.3±1.9 | 73.7±5.0/76.5±5.5 | 87.4±1.3 | 66.0±3.9/55.5±2.2 | 75.8±2.2 | 55.1±2.3/51.0±2.1 | 75.8±2.2 | 37.9±1.8/39.6±2.1 | 75.8±2.2 | 71.9±2.2/54.0±0.6 | 82.6±1.5 | 66.0±1.5/63.7±2.1 | 91.2±1.6 |
| | **GCN-GEBM** | 85.0±0.9/88.0±1.5 | 90.3±1.9 | 73.7±5.0/75.1±3.3 | 87.4±1.3 | 66.0±3.9/71.3±2.9 | 75.8±2.2 | 55.1±2.3/94.6±2.5 | 75.8±2.2 | 37.9±1.8/88.8±3.1 | 75.8±2.2 | 71.9±2.2/56.8±1.1 | 82.6±1.5 | 66.0±1.5/68.8±2.2 | 91.2±1.6 |
| CoraML-LLM | GCN-MCD | 84.0±1.7/46.4±2.3 | 90.1±1.0 | 74.0±5.4/32.1±2.3 | 89.5±2.3 | n.a./n.a. | n.a. | 70.3±3.3/52.9±2.3 | 76.9±1.4 | 41.1±1.5/47.8±1.7 | 76.9±1.4 | 70.0±1.3/55.3±1.8 | 84.0±1.5 | 65.4±0.9/49.6±2.3 | 91.0±0.7 |
| | GCN-DE | 80.7±1.9/58.2±2.3 | 87.8±1.2 | 72.2±4.8/45.7±3.1 | 87.9±2.2 | n.a./n.a. | n.a. | 75.1±2.8/37.9±0.7 | 75.6±1.5 | 39.9±1.3/64.8±2.3 | 75.6±1.5 | 65.0±1.4/67.8±1.7 | 81.8±2.2 | 64.8±1.4/51.3±2.1 | 88.5±1.1 |
| | GCN-MCD+DE | 80.3±2.1/58.3±2.1 | 88.1±1.4 | 72.6±6.0/46.3±3.5 | 87.9±1.9 | n.a./n.a. | n.a. | 75.5±2.3/37.8±1.1 | 75.2±2.1 | 39.7±1.4/64.6±2.3 | 75.2±2.1 | 64.7±0.9/69.3±1.4 | 81.3±1.8 | 64.8±2.0/51.7±1.5 | 88.7±1.0 |
| | GCN-BNN | 83.4±2.0/67.5±4.5 | 90.0±1.2 | 72.6±6.0/61.2±4.7 | 89.3±2.2 | n.a./n.a. | n.a. | 72.8±2.9/40.8±1.9 | 76.8±1.5 | 41.7±1.1/65.7±1.9 | 76.8±1.5 | 68.5±1.4/52.2±1.3 | 83.7±1.6 | 66.2±1.0/67.7±2.0 | 90.8±0.9 |
| | GCN-Ens | 83.9±1.8/70.1±5.3 | 90.1±1.0 | 74.1±5.2/57.5±6.9 | 89.5±2.4 | n.a./n.a. | n.a. | 70.3±3.4/46.2±3.5 | 77.3±1.3 | 39.6±1.5/65.6±1.6 | 77.3±1.3 | 70.2±1.1/56.9±1.3 | 84.2±1.2 | 65.4±0.6/62.2±2.4 | 91.3±0.6 |
| | GPN | 75.5±4.1/83.3±2.0 | 89.4±0.9 | 75.1±5.6/69.5±2.7 | 88.1±2.4 | n.a./n.a. | n.a. | 53.9±1.7/55.2±1.7 | 75.8±2.3 | 52.3±2.0/58.0±2.2 | 75.8±2.3 | 55.6±1.6/62.5±1.6 | 81.3±1.9 | 63.0±4.4/47.0±2.0 | 90.8±1.2 |
| | SGCN | 83.8±2.1/81.8±2.7 | 88.9±0.8 | 72.8±4.6/69.6±7.3 | 89.8±1.9 | n.a./n.a. | n.a. | 64.4±6.7/51.7±5.2 | 68.1±18.7 | 40.8±2.1/44.9±5.3 | 68.1±18.7 | 68.1±1.6/60.7±6.8 | 78.9±14.6 | 59.8±6.6/61.8±5.2 | 80.2±20.4 |
| | GCN-EBM | 83.7±1.9/84.0±2.3 | 90.2±0.9 | 74.1±5.1/75.9±6.5 | 89.4±2.3 | n.a./n.a. | n.a. | 70.5±3.0/65.1±2.7 | 77.1±1.3 | 39.9±1.5/39.2±1.5 | 77.1±1.3 | 69.9±1.2/69.1±1.8 | 83.7±1.5 | 65.3±0.7/66.5±1.0 | 91.1±0.7 |
| | GCN-HEAT | 83.7±1.9/81.1±4.2 | 90.2±0.9 | 74.1±5.1/68.0±8.4 | 89.4±2.3 | n.a./n.a. | n.a. | 70.5±3.0/69.1±7.0 | 77.1±1.3 | 39.9±1.5/67.0±4.7 | 77.1±1.3 | 69.9±1.2/58.6±3.3 | 83.7±1.5 | 65.3±0.7/62.7±3.5 | 91.1±0.7 |
| | GCNSafe | 83.7±1.9/86.4±1.7 | 90.2±0.9 | 74.1±5.1/80.1±4.5 | 89.4±2.3 | n.a./n.a. | n.a. | 70.5±3.0/57.4±1.1 | 77.1±1.3 | 39.9±1.5/43.6±2.2 | 77.1±1.3 | 69.9±1.2/54.8±0.7 | 83.7±1.5 | 65.3±0.7/62.7±1.2 | 91.1±0.7 |
| | **GCN-GEBM** | 83.7±1.9/84.7±2.0 | 90.2±0.9 | 74.1±5.1/76.8±5.1 | 89.4±2.3 | n.a./n.a. | n.a. | 70.5±3.0/72.8±4.9 | 77.1±1.3 | 39.9±1.5/87.3±3.8 | 77.1±1.3 | 69.9±1.2/53.2±0.6 | 83.7±1.5 | 65.3±0.7/67.5±2.1 | 91.1±0.7 |
| Citeseer | GCN-MCD | 70.4±4.2/31.6±2.2 | 84.8±1.2 | 78.2±3.6/38.3±2.0 | 90.9±0.6 | 61.8±1.9/50.7±2.7 | 76.4±1.5 | 48.1±2.4/48.6±2.2 | 76.4±1.5 | 36.8±1.6/48.4±2.5 | 76.4±1.5 | 60.1±2.2/50.1±2.1 | 78.4±1.1 | 51.3±1.7/50.6±1.6 | 77.8±1.9 |
| | GCN-DE | 61.7±4.9/42.4±2.5 | 81.0±1.2 | 72.5±3.5/51.6±3.2 | 86.4±1.0 | 59.2±2.4/54.5±2.0 | 72.9±1.6 | 45.2±2.6/79.9±2.4 | 72.9±1.6 | 35.1±1.5/65.8±2.7 | 72.9±1.6 | 56.0±1.8/56.1±1.6 | 75.6±1.4 | 51.5±1.5/50.4±1.5 | 74.3±1.8 |
| | GCN-MCD+DE | 63.1±5.1/43.4±2.7 | 81.3±1.2 | 73.1±3.0/52.5±2.0 | 87.1±1.1 | 58.2±2.5/55.8±2.7 | 73.0±1.5 | 46.1±3.5/80.1±2.4 | 73.0±1.5 | 34.9±1.3/65.4±2.5 | 73.0±1.5 | 56.0±1.9/56.6±1.1 | 75.0±1.8 | 51.3±1.7/50.7±1.1 | 74.6±1.6 |
| | GCN-BNN | 68.6±3.1/64.9±3.0 | 84.5±1.3 | 76.5±3.6/72.3±3.7 | 90.3±1.0 | 60.7±1.5/50.9±2.5 | 75.8±1.5 | 42.8±3.3/69.8±8.7 | 75.8±1.5 | 35.9±1.2/63.1±2.5 | 75.8±1.5 | 58.5±2.1/48.8±1.2 | 77.8±0.6 | 52.0±0.9/53.9±1.6 | 77.1±2.0 |
| | GCN-Ens | 71.3±3.9/52.3±4.4 | 84.8±1.0 | 78.4±3.8/63.5±2.8 | 91.1±0.5 | 61.6±1.4/54.1±2.8 | 76.9±1.6 | 48.7±1.5/77.9±3.4 | 76.9±1.6 | 36.5±1.1/59.1±3.4 | 76.9±1.6 | 60.2±2.3/43.7±0.9 | 78.6±1.0 | 51.2±1.7/57.8±1.3 | 77.7±1.8 |
| | GPN | 69.7±5.7/71.3±4.4 | 85.6±1.6 | 81.1±4.0/69.0±5.2 | 91.6±1.2 | 54.9±1.7/51.9±1.6 | 76.7±1.9 | 49.8±3.7/56.7±7.9 | 76.7±1.9 | 54.0±1.5/66.8±2.5 | 76.7±1.9 | 46.7±1.2/50.7±1.9 | 78.3±2.3 | 53.4±1.8/50.5±1.5 | 77.4±2.2 |
| | SGCN | 68.1±3.1/62.5±7.3 | 85.0±1.1 | 77.6±3.4/69.8±4.3 | 90.4±0.7 | 59.7±2.3/56.4±2.6 | 75.8±1.4 | 57.3±4.6/59.1±6.6 | 75.8±1.4 | 36.4±1.3/35.9±1.5 | 75.8±1.4 | 58.2±2.2/52.7±2.2 | 79.3±1.1 | 51.5±1.7/53.6±2.1 | 76.6±2.0 |
| | GCN-EBM | 71.0±3.9/71.0±3.8 | 84.8±1.0 | 78.4±3.9/76.3±4.3 | 90.9±0.7 | 61.9±3.1/59.0±2.4 | 76.6±1.5 | 49.8±4.8/42.5±3.3 | 76.6±1.5 | 36.8±1.4/34.8±1.4 | 76.6±1.5 | 60.1±2.2/59.1±2.1 | 78.7±1.1 | 51.3±1.7/51.8±2.1 | 77.5±1.9 |
| | GCN-HEAT | 71.0±3.9/63.2±4.2 | 84.8±1.0 | 78.4±3.9/71.9±4.1 | 90.9±0.7 | 61.9±3.1/58.3±2.2 | 76.6±1.5 | 49.8±4.8/73.9±3.2 | 76.6±1.5 | 36.8±1.4/77.0±2.7 | 76.6±1.5 | 60.1±2.2/55.4±1.9 | 78.7±1.1 | 51.3±1.7/51.7±1.5 | 77.5±1.9 |
| | GCNSafe | 71.0±3.9/73.8±4.3 | 84.8±1.0 | 78.4±3.9/77.3±5.3 | 90.9±0.7 | 61.9±3.1/54.6±1.6 | 76.6±1.5 | 49.8±4.8/44.0±2.4 | 76.6±1.5 | 36.8±1.4/37.9±1.4 | 76.6±1.5 | 60.1±2.2/45.3±1.5 | 78.7±1.1 | 51.3±1.7/53.4±1.8 | 77.5±1.9 |
| | **GCN-GEBM** | 71.0±3.9/69.4±4.5 | 84.8±1.0 | 78.4±3.9/78.5±3.0 | 90.9±0.7 | 61.9±3.1/59.3±2.0 | 76.6±1.5 | 49.8±4.8/96.6±2.7 | 76.6±1.5 | 36.8±1.4/75.3±3.9 | 76.6±1.5 | 60.1±2.2/47.2±1.4 | 78.7±1.1 | 51.3±1.7/52.4±1.8 | 77.5±1.9 |
| PubMed | GCN-MCD | 53.6±1.8/39.1±0.6 | 92.3±1.8 | 33.5±2.7/19.8±0.6 | 82.2±0.8 | n.a./n.a. | n.a. | 54.8±5.7/50.4±1.1 | 76.9±1.5 | 38.5±0.8/49.8±0.8 | 76.9±1.5 | 62.2±1.5/49.6±1.0 | 78.7±0.9 | 54.8±1.3/49.6±0.7 | 83.3±1.9 |
| | GCN-DE | 53.0±2.7/44.5±1.5 | 91.8±1.9 | 29.8±3.0/24.0±2.5 | 79.2±2.0 | n.a./n.a. | n.a. | 51.3±4.4/69.3±2.7 | 74.9±1.7 | 36.6±0.7/58.2±1.1 | 74.9±1.7 | 55.4±1.1/65.5±1.1 | 76.8±1.6 | 55.0±1.3/47.2±0.8 | 80.9±2.3 |
| | GCN-MCD+DE | 52.4±2.5/44.3±1.5 | 91.7±1.9 | 30.5±3.3/24.0±2.5 | 79.3±1.6 | n.a./n.a. | n.a. | 51.5±6.3/69.0±5.0 | 74.9±1.6 | 36.9±0.7/58.9±1.6 | 74.9±1.6 | 55.6±1.2/65.5±1.0 | 77.1±1.3 | 55.1±1.2/47.5±0.8 | 80.7±2.4 |
| | GCN-BNN | 52.5±3.6/50.3±3.7 | 92.2±1.7 | 32.8±2.6/31.5±4.0 | 81.7±3.1 | n.a./n.a. | n.a. | 50.5±5.3/60.9±3.6 | 76.4±1.1 | 37.9±0.7/64.2±2.3 | 76.4±1.1 | 60.3±2.1/46.4±2.5 | 77.1±3.3 | 54.2±1.6/59.3±2.0 | 82.8±2.2 |
| | GCN-Ens | 53.6±1.8/43.1±3.5 | 92.3±1.8 | 33.5±2.6/23.6±4.5 | 82.2±0.8 | n.a./n.a. | n.a. | 54.5±5.5/66.2±3.2 | 76.9±1.5 | 37.6±0.8/58.7±1.1 | 76.9±1.5 | 62.4±1.4/46.0±1.4 | 78.7±0.8 | 54.7±1.2/62.8±2.1 | 83.3±1.8 |
| | GPN | 50.1±3.4/53.5±2.7 | 92.9±0.8 | 34.0±4.5/39.4±5.1 | 85.0±1.3 | n.a./n.a. | n.a. | 54.7±3.4/51.9±3.8 | 76.9±1.1 | 54.1±1.6/73.0±3.7 | 76.9±1.1 | 47.7±0.6/66.1±3.3 | 80.2±1.1 | 56.9±2.8/45.7±1.5 | 86.2±1.5 |
| | SGCN | 52.4±2.0/56.6±2.4 | 92.7±1.1 | 32.4±2.8/35.4±5.0 | 83.8±1.2 | n.a./n.a. | n.a. | 58.4±4.6/52.2±5.9 | 77.2±1.4 | 38.7±0.6/39.5±0.9 | 77.2±1.4 | 62.4±1.2/70.2±3.5 | 80.8±1.0 | 55.1±1.5/54.0±3.8 | 83.7±1.9 |
| | GCN-EBM | 53.6±1.8/53.5±2.1 | 92.3±1.8 | 33.7±2.8/33.0±3.3 | 82.2±0.8 | n.a./n.a. | n.a. | 54.7±5.8/46.1±3.2 | 76.9±1.4 | 38.2±0.6/39.1±0.7 | 76.9±1.4 | 62.3±1.5/66.0±3.2 | 78.6±0.9 | 54.8±1.3/53.7±1.7 | 83.3±2.0 |
| | GCN-HEAT | 53.6±1.8/50.9±1.9 | 92.3±1.8 | 33.7±2.8/28.3±4.1 | 82.2±0.8 | n.a./n.a. | n.a. | 54.7±5.8/50.1±5.4 | 76.9±1.4 | 38.2±0.6/75.3±7.9 | 76.9±1.4 | 62.3±1.5/51.9±5.6 | 78.6±0.9 | 54.8±1.3/57.9±2.6 | 83.3±2.0 |
| | GCNSafe | 53.6±1.8/54.3±2.9 | 92.3±1.8 | 33.7±2.8/40.5±5.0 | 82.2±0.8 | n.a./n.a. | n.a. | 54.7±5.8/48.4±1.9 | 76.9±1.4 | 38.2±0.6/42.5±0.6 | 76.9±1.4 | 62.3±1.5/50.7±1.4 | 78.6±0.9 | 54.8±1.3/55.8±1.2 | 83.3±2.0 |
| | **GCN-GEBM** | 53.6±1.8/56.7±1.7 | 92.3±1.8 | 33.7±2.8/33.4±6.1 | 82.2±0.8 | n.a./n.a. | n.a. | 54.7±5.8/83.8±8.8 | 76.9±1.4 | 38.2±0.6/79.8±3.2 | 76.9±1.4 | 62.3±1.5/49.0±1.2 | 78.6±0.9 | 54.8±1.3/56.1±1.8 | 83.3±2.0 |
| Amazon Computers | GCN-MCD | 50.2±8.4/29.1±1.5 | 88.8±1.8 | 81.2±5.3/42.2±1.6 | 94.2±1.1 | 57.5±1.2/51.0±1.6 | 80.2±2.5 | 54.4±2.9/50.4±1.4 | 80.2±2.5 | 44.2±1.4/49.1±1.6 | 80.2±2.5 | 80.0±1.2/54.6±1.3 | 81.0±1.5 | 59.7±0.7/52.0±0.7 | 91.2±1.3 |
| | GCN-DE | 52.5±9.0/42.0±2.1 | 87.9±1.9 | 82.0±4.1/60.0±1.7 | 93.4±1.3 | 57.6±1.5/52.1±1.7 | 78.8±2.5 | 55.2±3.2/51.3±1.6 | 78.8±2.5 | 44.3±1.7/56.0±1.3 | 78.8±2.5 | 77.5±1.3/69.7±0.6 | 80.4±1.3 | 62.2±0.7/49.3±0.3 | 89.5±1.4 |
| | GCN-MCD+DE | 52.3±8.0/41.7±2.0 | 88.2±1.7 | 81.3±4.9/59.7±1.4 | 93.4±1.4 | 57.9±1.3/51.9±1.5 | 78.5±2.8 | 55.1±2.7/51.4±1.7 | 78.5±2.8 | 44.4±1.7/56.0±1.7 | 78.5±2.8 | 77.3±1.5/69.7±0.6 | 80.5±1.2 | 62.0±0.9/49.2±0.4 | 89.8±1.4 |
| | GCN-BNN | 50.4±11.6/54.0±10.8 | 84.3±9.1 | 77.9±5.3/76.8±6.1 | 93.4±1.9 | 56.7±1.3/52.5±1.5 | 78.3±2.5 | 53.6±2.5/52.4±1.9 | 78.3±2.5 | 44.1±1.4/57.1±1.5 | 78.3±2.5 | 76.2±1.9/45.1±2.9 | 77.7±3.2 | 59.4±1.4/64.7±2.0 | 90.8±1.4 |
| | GCN-Ens | 50.5±8.7/50.2±5.5 | 88.9±1.9 | 81.0±5.5/70.3±5.2 | 94.2±1.1 | 57.6±1.3/48.8±1.6 | 80.4±2.5 | 54.5±3.1/51.6±2.1 | 80.4±2.5 | 43.7±1.2/58.0±1.9 | 80.4±2.5 | 80.2±1.1/52.4±1.6 | 81.2±1.3 | 59.7±0.5/59.6±1.4 | 91.3±1.2 |
| | GPN | 56.3±10.2/47.8±4.3 | 86.4±1.9 | 68.1±10.4/69.0±2.9 | 94.3±1.1 | 53.8±2.3/55.0±1.4 | 74.4±14.7 | 54.8±2.5/56.4±1.5 | 74.4±14.7 | 58.8±9.4/59.7±4.1 | 74.4±14.7 | 51.1±0.8/71.4±1.4 | 78.5±2.2 | 59.5±3.6/47.0±1.2 | 92.4±1.2 |
| | SGCN | 26.6±7.3/36.6±5.7 | 39.5±27.2 | 49.3±16.0/50.8±19.4 | 59.1±26.4 | 48.5±0.9/49.9±1.5 | 11.7±12.2 | 47.7±0.9/50.5±1.6 | 11.7±12.2 | 48.7±2.2/48.5±1.8 | 11.7±12.2 | 93.3±0.8/32.4±6.5 | 20.0±14.3 | 45.5±0.3/59.4±0.1 | 10.2±9.5 |
| | GCN-EBM | 50.8±8.8/53.7±7.3 | 88.8±1.9 | 80.7±5.7/79.1±4.5 | 94.1±1.2 | 57.4±1.3/56.6±1.1 | 80.0±2.6 | 54.7±3.0/54.2±3.0 | 80.0±2.6 | 44.1±1.3/43.5±1.2 | 80.0±2.6 | 79.9±1.1/81.8±1.8 | 81.0±1.7 | 59.6±0.6/60.0±1.1 | 91.1±1.2 |
| | GCN-HEAT | 50.8±8.8/49.6±7.9 | 88.8±1.9 | 80.7±5.7/73.0±5.8 | 94.1±1.2 | 57.4±1.3/54.4±1.7 | 80.0±2.6 | 54.7±3.0/53.4±3.1 | 80.0±2.6 | 44.1±1.3/61.2±2.0 | 80.0±2.6 | 79.9±1.1/71.1±4.7 | 81.0±1.7 | 59.6±0.6/62.0±1.9 | 91.1±1.2 |
| | GCNSafe | 50.8±8.8/61.4±4.3 | 88.8±1.9 | 80.7±5.7/71.3±3.7 | 94.1±1.2 | 57.4±1.3/52.7±1.5 | 80.0±2.6 | 54.7±3.0/51.9±2.3 | 80.0±2.6 | 44.1±1.3/44.2±1.3 | 80.0±2.6 | 79.9±1.1/54.1±0.5 | 81.0±1.7 | 59.6±0.6/52.3±0.6 | 91.1±1.2 |
| | **GCN-GEBM** | 50.8±8.8/65.0±5.3 | 88.8±1.9 | 80.7±5.7/77.6±6.2 | 94.1±1.2 | 57.4±1.3/79.5±3.1 | 80.0±2.6 | 54.7±3.0/67.4±7.1 | 80.0±2.6 | 44.1±1.3/96.0±4.6 | 80.0±2.6 | 79.9±1.1/50.1±1.2 | 81.0±1.7 | 59.6±0.6/61.3±1.2 | 91.1±1.2 |
| Amazon Photo | GCN-MCD | 62.1±6.3/41.6±1.2 | 92.0±1.1 | 55.8±3.4/28.8±1.7 | 95.3±2.0 | 58.8±0.9/50.9±1.5 | 89.8±1.1 | 52.1±1.9/50.3±1.9 | 89.8±1.1 | 45.2±1.0/48.7±1.5 | 89.8±1.1 | 82.2±1.5/52.7±1.3 | 91.9±0.5 | 63.7±0.6/51.8±1.5 | 97.4±0.2 |
| | GCN-DE | 65.0±5.5/46.1±1.7 | 91.7±1.1 | 57.6±3.6/40.6±2.1 | 94.6±2.0 | 59.5±0.9/53.7±1.0 | 88.5±1.2 | 53.1±1.8/51.4±1.4 | 88.5±1.2 | 44.5±0.9/54.1±1.6 | 88.5±1.2 | 80.4±1.1/75.7±0.5 | 91.3±0.7 | 64.9±1.0/49.1±0.7 | 96.6±0.4 |
| | GCN-MCD+DE | 64.9±5.2/46.0±1.7 | 91.4±1.1 | 57.2±4.6/40.1±1.9 | 94.8±1.9 | 59.1±1.0/53.4±1.0 | 88.3±1.8 | 52.6±2.1/51.0±1.0 | 88.3±1.8 | 44.7±1.2/54.3±1.2 | 88.3±1.8 | 80.4±1.5/75.7±0.7 | 91.6±0.7 | 64.6±1.0/49.2±0.6 | 96.6±0.4 |
| | GCN-BNN | 59.9±6.4/62.9±6.6 | 91.6±2.0 | 62.1±5.3/71.9±7.0 | 95.7±1.9 | 56.4±1.4/52.4±1.6 | 89.3±2.1 | 50.0±1.6/48.9±1.6 | 89.3±2.1 | 45.4±0.8/57.5±1.8 | 89.3±2.1 | 74.6±2.5/57.3±4.4 | 90.4±2.7 | 64.9±2.8/68.1±3.9 | 97.6±0.4 |
| | GCN-Ens | 62.1±6.2/69.1±8.7 | 91.9±1.0 | 55.9±3.2/62.5±10.2 | 95.3±1.9 | 58.7±0.7/51.8±1.3 | 89.9±1.0 | 52.4±2.3/51.2±1.0 | 89.9±1.0 | 44.5±1.6/55.3±1.7 | 89.9±1.0 | 82.4±1.5/66.3±2.0 | 92.1±0.5 | 63.6±0.5/64.6±1.8 | 97.5±0.2 |
| | GPN | 58.4±6.1/80.6±3.2 | 88.8±1.3 | 67.1±8.9/44.8±5.4 | 93.9±2.2 | 54.5±1.5/54.1±1.4 | 86.5±1.8 | 54.5±1.4/54.2±1.5 | 86.5±1.8 | 55.2±1.4/55.8±1.7 | 86.5±1.8 | 51.7±0.9/71.4±3.1 | 88.6±2.0 | 65.5±3.3/46.0±2.0 | 97.1±0.5 |
| | SGCN | 63.5±3.7/65.1±6.6 | 92.1±0.9 | 50.3±3.1/51.6±8.2 | 94.3±2.2 | 48.7±0.4/50.2±1.5 | 15.4±10.0 | 47.9±0.6/50.4±1.9 | 15.4±10.0 | 48.3±2.0/49.3±2.3 | 15.4±10.0 | 92.6±1.7/42.6±22.6 | 12.2±8.6 | 46.1±1.3/57.2±2.9 | 15.0±5.9 |
| | GCN-EBM | 62.1±6.4/63.3±7.7 | 91.8±1.0 | 55.9±3.2/51.4±4.2 | 95.3±2.0 | 58.8±0.8/59.0±0.7 | 89.8±1.0 | 52.3±2.1/52.0±1.7 | 89.8±1.0 | 45.8±0.8/45.1±0.7 | 89.8±1.0 | 82.4±1.6/89.0±1.4 | 91.9±0.6 | 63.5±0.5/62.1±1.3 | 97.4±0.3 |
| | GCN-HEAT | 62.1±6.4/63.2±5.7 | 91.8±1.0 | 55.9±3.2/45.9±5.5 | 95.3±2.0 | 58.8±0.8/55.9±1.9 | 89.8±1.0 | 52.3±2.1/51.1±2.0 | 89.8±1.0 | 45.8±0.8/59.9±2.0 | 89.8±1.0 | 82.4±1.6/72.9±4.7 | 91.9±0.6 | 63.5±0.5/63.0±2.1 | 97.4±0.3 |
| | GCNSafe | 62.1±6.4/60.1±9.1 | 91.8±1.0 | 55.9±3.2/56.4±5.8 | 95.3±2.0 | 58.8±0.8/53.8±1.6 | 89.8±1.0 | 52.3±2.1/50.5±1.5 | 89.8±1.0 | 45.8±0.8/46.6±0.5 | 89.8±1.0 | 82.4±1.6/59.7±0.9 | 91.9±0.6 | 63.5±0.5/55.0±0.9 | 97.4±0.3 |
| | **GCN-GEBM** | 62.1±6.4/74.9±8.6 | 91.8±1.0 | 55.9±3.2/62.1±7.3 | 95.3±2.0 | 58.8±0.8/77.0±4.0 | 89.8±1.0 | 52.3±2.1/55.7±6.4 | 89.8±1.0 | 45.8±0.8/94.8±1.9 | 89.8±1.0 | 82.4±1.6/55.5±0.8 | 91.9±0.6 | 63.5±0.5/66.4±1.8 | 97.4±0.3 |
| Coauthor-CS | GCN-MCD | 89.0±1.6/56.6±1.4 | 92.3±0.8 | 76.8±2.4/25.2±2.0 | 95.1±0.5 | 68.5±0.9/51.9±1.0 | 89.9±1.0 | 69.4±1.7/52.2±0.9 | 89.9±1.0 | 39.4±0.3/48.0±0.6 | 89.9±1.0 | 72.6±0.9/53.8±0.9 | 90.8±0.4 | 63.5±0.5/51.2±1.0 | 97.9±0.4 |
| | GCN-DE | 89.3±1.3/67.2±1.4 | 91.3±0.7 | 76.6±2.4/37.4±1.8 | 94.4±0.7 | 66.6±0.6/59.3±0.8 | 88.8±1.1 | 60.0±1.9/68.7±1.1 | 88.8±1.1 | 39.2±0.5/61.4±0.7 | 88.8±1.1 | 67.1±1.0/75.3±0.9 | 89.6±0.6 | 62.5±0.6/52.1±0.6 | 97.1±0.5 |
| | GCN-MCD+DE | 89.1±1.6/67.0±1.5 | 91.4±0.5 | 75.3±3.0/36.9±2.1 | 94.3±0.6 | 66.5±0.9/59.4±0.8 | 88.5±1.0 | 60.3±1.8/68.8±0.8 | 88.5±1.0 | 39.2±0.5/60.7±0.7 | 88.5±1.0 | 66.8±1.1/75.0±1.0 | 89.6±0.6 | 62.5±0.7/52.1±0.5 | 97.1±0.4 |
| | GCN-BNN | 82.9±2.6/80.8±2.7 | 90.8±0.8 | 67.5±3.0/58.5±5.3 | 94.7±0.5 | 66.2±1.0/63.4±1.1 | 89.0±0.8 | 51.1±2.5/68.8±2.3 | 89.0±0.8 | 36.8±1.0/55.3±1.6 | 89.0±0.8 | 65.7±1.0/54.2±0.8 | 89.8±0.6 | 67.9±0.8/72.6±0.9 | 97.9±0.4 |
| | GCN-Ens | 89.3±1.4/83.9±3.4 | 92.4±0.8 | 76.9±2.0/61.0±3.8 | 95.2±0.5 | 68.6±1.1/58.3±1.6 | 90.0±0.9 | 69.3±1.7/64.3±1.8 | 90.0±0.9 | 38.6±0.6/61.3±0.8 | 90.0±0.9 | 72.6±0.9/65.5±0.7 | 90.8±0.4 | 63.5±0.3/71.2±0.5 | 97.9±0.4 |
| | GPN | 68.5±3.8/88.7±1.2 | 86.6±1.4 | 53.7±5.3/44.5±6.7 | 91.6±1.2 | 54.9±1.3/56.8±1.4 | 81.7±2.3 | 52.9±1.0/59.3±2.4 | 81.7±2.3 | 52.6±1.0/59.7±2.5 | 81.7±2.3 | 50.2±0.5/77.5±2.1 | 83.4±1.6 | 57.0±3.2/43.2±0.6 | 97.9±0.4 |
| | SGCN | 58.5±6.4/42.9±1.2 | 30.6±14.7 | 29.4±11.4/10.0±0.7 | 31.2±16.2 | 46.3±0.8/49.7±0.6 | 11.0±7.0 | 38.7±1.0/59.9±0.7 | 11.0±7.0 | 39.3±1.2/52.1±2.5 | 11.0±7.0 | 80.2±3.9/34.0±0.6 | 12.1±9.1 | 41.0±1.5/61.5±1.0 | 14.1±6.5 |
| | GCN-EBM | 89.2±1.5/89.6±1.8 | 92.3±0.9 | 76.5±2.4/78.7±3.3 | 95.1±0.5 | 68.5±1.1/69.4±1.0 | 89.8±0.9 | 68.8±1.8/71.9±2.4 | 89.8±0.9 | 39.5±0.5/37.2±0.5 | 89.8±0.9 | 72.6±0.9/77.8±1.2 | 90.9±0.3 | 63.5±0.4/60.9±0.8 | 97.8±0.4 |
| | GCN-HEAT | 89.2±1.5/85.2±2.4 | 92.3±0.9 | 76.5±2.4/57.8±7.0 | 95.1±0.5 | 68.5±1.1/67.3±1.6 | 89.8±0.9 | 68.8±1.8/65.5±1.9 | 89.8±0.9 | 39.5±0.5/69.4±0.5 | 89.8±0.9 | 72.6±0.9/64.6±3.4 | 90.9±0.3 | 63.5±0.4/65.8±1.3 | 97.8±0.4 |
| | GCNSafe | 89.2±1.5/92.6±1.5 | 92.3±0.9 | 76.5±2.4/84.8±2.1 | 95.1±0.5 | 68.5±1.1/61.1±0.6 | 89.8±0.9 | 68.8±1.8/62.2±1.6 | 89.8±0.9 | 39.5±0.5/41.0±0.6 | 89.8±0.9 | 72.6±0.9/61.3±0.7 | 90.9±0.3 | 63.5±0.4/58.0±0.6 | 97.8±0.4 |
| | **GCN-GEBM** | 89.2±1.5/93.4±1.2 | 92.3±0.9 | 76.5±2.4/81.6±5.1 | 95.1±0.5 | 68.5±1.1/93.9±1.4 | 89.8±0.9 | 68.8±1.8/99.6±0.2 | 89.8±0.9 | 39.5±0.5/92.8±3.9 | 89.8±0.9 | 72.6±0.9/55.2±0.8 | 90.9±0.3 | 63.5±0.4/65.7±1.0 | 97.8±0.4 |
| Coauthor-Physics | GCN-MCD | 76.0±2.4/18.4±0.7 | 96.7±0.5 | 89.7±1.7/25.6±0.7 | 98.3±0.2 | 59.1±1.0/50.7±0.6 | 91.9±1.2 | 62.4±1.0/50.8±1.0 | 91.9±1.2 | 45.0±1.0/49.7±0.8 | 91.9±1.2 | 73.1±1.3/49.6±0.5 | 93.4±1.0 | 59.0±0.4/49.8±0.6 | 98.0±0.2 |
| | GCN-DE | 76.5±3.2/48.7±2.8 | 96.4±0.7 | 88.6±1.8/69.9±1.5 | 98.1±0.2 | 58.7±0.9/55.6±0.7 | 91.1±1.2 | 61.5±1.7/60.0±0.6 | 91.1±1.2 | 44.7±1.0/53.6±1.0 | 91.1±1.2 | 68.7±1.1/74.2±0.9 | 93.1±1.0 | 58.9±0.6/53.4±0.4 | 97.7±0.2 |
| | GCN-MCD+DE | 76.6±3.0/49.4±2.4 | 96.5±0.6 | 89.0±1.8/70.2±2.1 | 98.2±0.2 | 58.5±0.6/55.5±0.6 | 91.3±1.5 | 61.4±1.6/59.8±0.6 | 91.3±1.5 | 44.7±1.0/53.6±0.8 | 91.3±1.5 | 68.9±1.1/74.4±0.8 | 93.1±0.9 | 58.8±0.5/53.5±0.6 | 97.7±0.2 |
| | GCN-BNN | 66.6±7.4/59.8±10.3 | 96.6±1.1 | 81.3±5.4/81.9±5.9 | 98.1±0.3 | 58.9±1.0/57.6±1.1 | 91.7±1.4 | 52.7±2.3/62.3±3.7 | 91.7±1.4 | 40.5±0.8/56.1±2.5 | 91.7±1.4 | 68.6±1.2/57.2±1.7 | 93.4±0.9 | 61.9±1.1/65.3±1.7 | 98.4±0.2 |
| | GCN-Ens | 76.0±2.4/60.0±6.0 | 96.8±0.5 | 89.8±1.4/79.5±3.1 | 98.3±0.2 | 59.2±1.0/55.3±1.3 | 92.0±1.2 | 62.4±1.0/57.2±1.2 | 92.0±1.2 | 44.3±0.9/54.6±1.1 | 92.0±1.2 | 73.2±1.2/62.0±2.8 | 93.3±1.0 | 59.0±0.4/66.1±1.2 | 98.0±0.2 |
| | GPN | 59.3±5.1/74.9±4.3 | 96.8±0.4 | 55.6±5.5/84.2±2.7 | 97.7±0.3 | 56.0±1.2/56.1±0.9 | 90.3±1.4 | 54.0±0.9/58.8±1.7 | 90.3±1.4 | 53.6±0.8/59.2±1.7 | 90.3±1.4 | 51.4±0.7/77.6±2.2 | 92.3±1.0 | 59.7±5.7/46.8±1.5 | 98.8±0.2 |
| | SGCN | 75.0±3.4/76.3±6.8 | 96.9±0.4 | 89.8±1.6/89.5±3.4 | 98.3±0.2 | 59.8±0.7/60.6±1.6 | 91.3±1.4 | 61.5±1.4/60.6±3.4 | 91.3±1.4 | 44.7±0.9/44.4±1.1 | 91.3±1.4 | 72.6±1.3/74.5±4.3 | 93.3±0.7 | 58.6±1.0/57.1±2.5 | 97.5±0.6 |
| | GCN-EBM | 76.0±2.5/79.8±3.2 | 96.8±0.5 | 89.7±1.3/92.0±1.4 | 98.3±0.2 | 59.0±1.0/58.8±1.2 | 92.0±1.2 | 62.3±1.1/62.1±1.9 | 92.0±1.2 | 44.8±1.0/43.6±0.9 | 92.0±1.2 | 73.1±1.3/77.3±2.0 | 93.4±1.0 | 59.0±0.4/56.9±0.8 | 98.0±0.2 |
| | GCN-HEAT | 76.0±2.5/63.9±3.2 | 96.8±0.5 | 89.7±1.3/78.0±2.0 | 98.3±0.2 | 59.0±1.0/56.6±1.1 | 92.0±1.2 | 62.3±1.1/61.6±1.6 | 92.0±1.2 | 44.8±1.0/61.0±0.8 | 92.0±1.2 | 73.1±1.3/66.9±2.3 | 93.4±1.0 | 59.0±0.4/58.8±0.6 | 98.0±0.2 |
| | GCNSafe | 76.0±2.5/86.3±2.8 | 96.8±0.5 | 89.7±1.3/94.4±0.9 | 98.3±0.2 | 59.0±1.0/54.9±0.8 | 92.0±1.2 | 62.3±1.1/56.4±1.1 | 92.0±1.2 | 44.8±1.0/46.1±0.8 | 92.0±1.2 | 73.1±1.3/62.5±1.3 | 93.4±1.0 | 59.0±0.4/56.0±0.8 | 98.0±0.2 |
| | **GCN-GEBM** | 76.0±2.5/84.7±3.8 | 96.8±0.5 | 89.7±1.3/95.4±0.8 | 98.3±0.2 | 59.0±1.0/77.1±2.7 | 92.0±1.2 | 62.3±1.1/94.1±1.9 | 92.0±1.2 | 44.8±1.0/80.1±4.4 | 92.0±1.2 | 73.1±1.3/55.9±1.0 | 93.4±1.0 | 59.0±0.4/61.3±0.8 | 98.0±0.2 |

Table 10: Out-of-Distribution detection AUC-PR (↑) using aleatoric or epistemic uncertainty (best and runner-up) in an inductive setting.

**Structural Anomalies.** The last anomaly type we consider is creating structural anomalies: To that end, we generate synthetic CSBM graphs similar to the node anomaly experiment. However, we adapt the intra- and inter-class edge probabilities to accommodate varying degrees of homophily: We fix the intra-class connection probability at $p = 0.05$ and vary the signal-to-noise ratio $\sigma_{\text{SNR}}$ which implicitly defines the inter-class edge probability as $p/\sigma_{\text{SNR}}$. We measure the degree of heterophily as $-\log \sigma_{\text{SNR}}$ and again compare how individual node energies change for increasingly heterophilic graphs in Figure 4c. We find that the cluster-level energy of GEBM is the only energy term that is sensitive to structural anomalies.

Overall, these experiments motivate the use of the three naturally arising energy terms in GEBM: Each energy is sensitive to a different family of distribution shifts that originate from anomalies at different structural scales on the graph: At the node level, with the local structure of a node and at the cluster-level.

### C.8 Overconfidence of GNNs

We disambiguate between previous studies on the underconfidence of GNNs [74, 75] and the overconfidence issue of piecewise affine networks tackled by our work. The former studies the calibration

| **Model** | CoraML | CoraML LLM | Citeseer | PubMed | Amazon Computers | Amazon Photo | Coauthor CS | Coauthor Physics |
|---|---|---|---|---|---|---|---|---|
| GCN-MCD | 9.7/18.9 | 9.2/19.0 | 9.4/18.7 | 9.1/18.7 | 9.1/18.3 | 9.9/19.1 | 10.0/19.2 | 10.5/20.4 |
| GCN-DE | 6.3/14.1 | 7.9/16.9 | 7.2/15.3 | 7.1/13.3 | 7.4/16.4 | 6.9/15.9 | 6.3/13.2 | 7.8/15.4 |
| GCN-MCD+DE | 6.8/14.7 | 7.2/15.9 | 6.0/13.9 | 6.6/12.8 | 7.9/16.9 | 7.6/16.6 | 6.7/13.5 | 7.4/15.1 |
| GCN-BNN | 5.9/12.3 | 6.6/14.1 | 5.6/11.8 | 5.8/11.4 | 4.6/10.4 | 5.3/9.7 | 5.2/11.6 | 5.3/11.9 |
| GCN-Ens | 7.4/15.7 | 6.8/16.0 | 6.8/13.5 | 7.2/13.4 | 6.4/14.9 | **4.4**/**8.9** | 4.9/11.1 | 6.4/13.6 |
| GPN | 7.1/15.3 | 6.0/14.8 | 6.4/13.1 | 4.8/9.0 | 5.4/12.9 | 5.3/11.8 | 6.8/13.5 | 5.6/12.5 |
| SGCN | 4.7/9.1 | 5.9/14.8 | 5.8/13.1 | **4.2**/**7.4** | 9.4/18.7 | 7.4/14.8 | 9.4/18.6 | 4.6/9.9 |
| GCN-EBM | **4.2**/**8.6** | **3.8**/**7.6** | 4.8/11.9 | 6.5/12.5 | **3.4**/**8.7** | 4.6/10.7 | **3.8**/**6.6** | **4.2**/**8.7** |
| GCN-HEAT | 4.8/11.8 | 4.6/13.1 | **4.7**/**11.0** | 5.5/12.7 | 3.6/10.9 | 4.7/12.4 | 4.6/10.6 | 5.2/13.7 |
| GCNSafe | 6.4/11.9 | 4.8/10.1 | 5.6/11.2 | 5.4/8.9 | 5.8/13.3 | 6.9/14.7 | 5.4/10.3 | 6.1/11.9 |
| **GCN-GEBM** | **2.7 / 4.5** | **3.2 / 5.7** | **3.6 / 7.7** | **3.6 / 7.0** | **2.8 / 6.5** | **3.0 / 5.0** | **2.8 / 4.7** | **3.0 / 4.8** |

Table 11: Average o.o.d. detection rank (AUC-PR) (↓) of epistemic uncertainty versus other epistemic measures / all uncertainty measures over all distribution shifts in an inductive setting (best and runner-up).

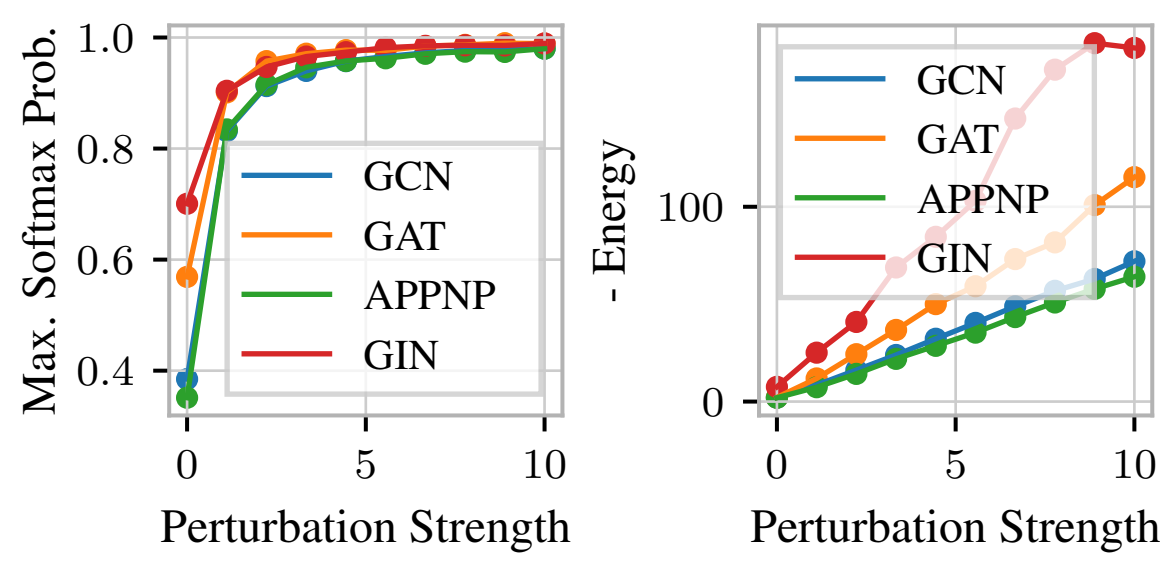

Figure 5: Confidence (Maximum Softmax Probability and negative energy) for different GNNs at increasing distribution shift severity.

of GNNs on in-distribution data, and provides strong evidence for models to be underconfident. This does not conflict with our claim that with increasing distance from the training distribution, i.e. on out-of-distribution data, piece affine GNNs become overconfident. We empirically verify this in Figure 5: Both the energy and the maximum softmax probability - confidence measures - of the GNN architectures visualized increase (or converge to their maximum) under more severe normal feature perturbations.

### C.9 Robust Evidential Inference

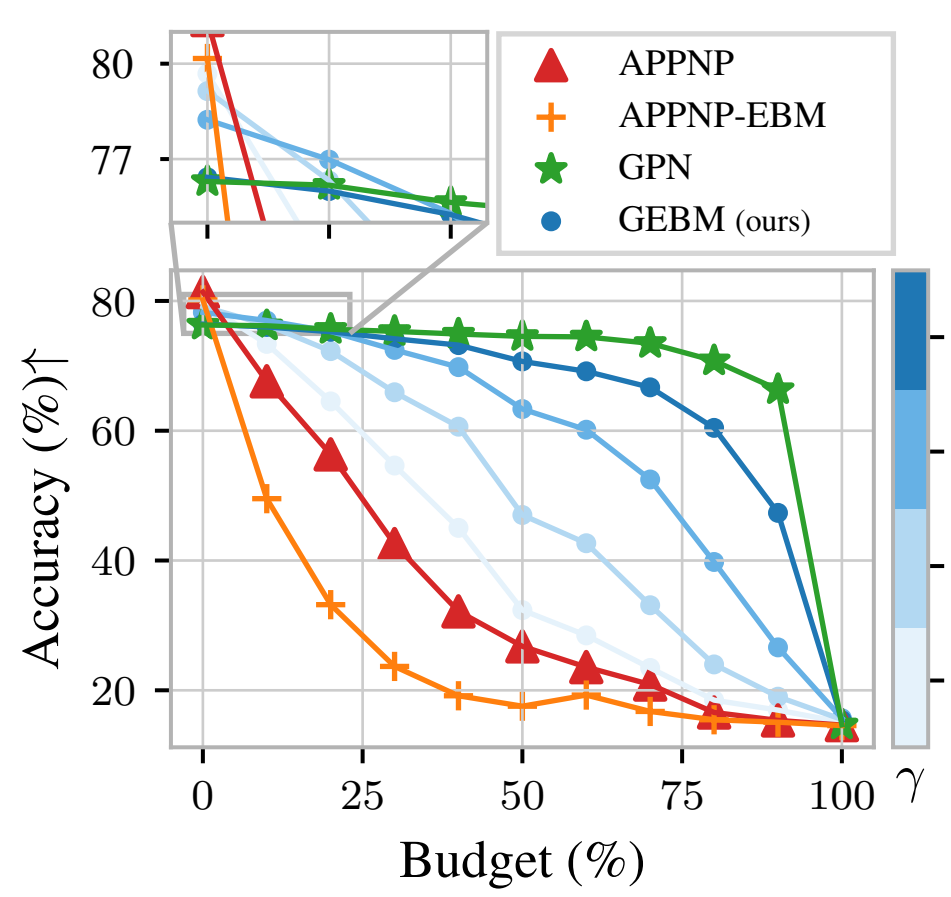

Figure 6: Robust evidential inference using APPNP as a backbone for GEBM at increasingly severy feature perturbations.

| Dataset | Model | LoC (last) AUC-PR↑ (Alea. / Epi.) | LoC (last) Acc.↑ | LoC (hetero.) AUC-PR↑ (Alea. / Epi.) | LoC (hetero.) Acc.↑ | Ber($\hat{p}$) (near) AUC-PR↑ (Alea. / Epi.) | Ber($\hat{p}$) (near) Acc.↑ |
|---|---|---|---|---|---|---|---|
| CoraML | GCN-MCD | 82.8±2.4/46.4±2.6 | 90.4±0.9 | 69.5±3.1/33.4±2.3 | 85.2±3.3 | 65.3±2.5/51.9±3.9 | 82.0±0.8 |
| | GCN-DE | 79.4±2.5/61.2±2.5 | 87.3±1.2 | 64.2±4.7/41.5±3.1 | 80.9±2.3 | 63.6±3.2/55.9±3.9 | 78.4±1.9 |
| | GCN-MCD+DE | 79.4±2.8/62.2±2.2 | 87.6±1.2 | 63.3±5.3/41.2±3.0 | 80.1±2.9 | 63.9±3.5/55.8±3.7 | 78.1±2.2 |
| | GCN-BNN | 77.2±4.1/77.4±3.6 | 89.0±1.2 | 58.2±5.3/54.3±6.0 | 82.1±3.4 | 64.1±2.8/58.4±3.3 | 80.6±2.0 |
| | GCN-Ens | 82.8±2.3/68.0±2.8 | 90.3±0.9 | 69.5±3.3/42.4±6.4 | 85.2±3.3 | 66.3±2.4/51.3±2.5 | 81.8±0.8 |
| | GPN | 76.6±2.2/82.3±1.5 | 90.0±1.0 | 65.0±8.1/63.2±4.2 | 85.5±2.2 | 50.8±3.2/53.1±4.2 | 78.8±2.1 |
| | SGCN | 83.1±1.7/86.1±2.5 | 91.0±1.2 | 73.2±3.2/78.9±3.1 | 88.5±2.0 | 54.9±9.6/54.5±5.9 | 51.7±30.2 |
| | GCN-EBM | 82.8±2.2/83.0±2.1 | 90.4±0.9 | 69.1±3.5/69.6±4.3 | 85.0±3.2 | 66.4±3.1/66.1±3.1 | 81.9±0.8 |
| | GCN-HEAT | 82.8±2.2/79.4±3.3 | 90.4±0.9 | 69.1±3.5/63.0±7.3 | 85.0±3.2 | 66.4±3.1/62.9±4.2 | 81.9±0.8 |
| | GCNSafe | 82.8±2.2/85.3±2.7 | 90.4±0.9 | 69.1±3.5/73.2±4.5 | 85.0±3.2 | 66.4±3.1/57.7±2.6 | 81.9±0.8 |
| | **GCN-GEBM** | 82.8±2.2/86.1±1.6 | 90.4±0.9 | 69.1±3.5/73.2±2.8 | 85.0±3.2 | 66.4±3.1/71.2±4.4 | 81.9±0.8 |
| CoraML-LLM | GCN-MCD | 84.4±3.5/45.7±2.2 | 91.2±0.7 | 70.7±4.0/33.2±1.8 | 90.6±2.0 | n.a./n.a. | n.a. |
| | GCN-DE | 82.8±3.4/61.5±2.6 | 89.7±1.2 | 69.1±5.3/49.2±3.4 | 88.9±1.7 | n.a./n.a. | n.a. |
| | GCN-MCD+DE | 83.2±2.4/61.3±2.4 | 89.4±0.9 | 68.7±4.8/48.8±3.4 | 88.9±1.7 | n.a./n.a. | n.a. |
| | GCN-BNN | 83.8±3.1/72.5±4.0 | 91.4±0.8 | 69.9±4.4/61.9±4.3 | 91.0±2.0 | n.a./n.a. | n.a. |
| | GCN-Ens | 84.5±3.5/78.1±2.7 | 91.4±0.7 | 70.9±3.8/56.6±6.1 | 90.6±2.3 | n.a./n.a. | n.a. |
| | GPN | 79.6±2.3/85.4±1.8 | 90.5±1.1 | 73.4±9.3/70.9±3.8 | 88.5±2.2 | n.a./n.a. | n.a. |
| | SGCN | 86.2±2.7/88.7±1.9 | 90.3±0.9 | 74.5±3.9/79.3±5.0 | 90.6±1.6 | n.a./n.a. | n.a. |
| | GCN-EBM | 84.3±3.5/84.0±4.7 | 91.4±0.7 | 70.9±3.8/73.0±4.5 | 90.7±2.0 | n.a./n.a. | n.a. |
| | GCN-HEAT | 84.3±3.5/78.0±7.0 | 91.4±0.7 | 70.9±3.8/65.4±8.0 | 90.7±2.0 | n.a./n.a. | n.a. |
| | GCNSafe | 84.3±3.5/86.7±4.8 | 91.4±0.7 | 70.9±3.8/79.8±5.0 | 90.7±2.0 | n.a./n.a. | n.a. |
| | **GCN-GEBM** | 84.3±3.5/85.9±2.9 | 91.4±0.7 | 70.9±3.8/74.6±6.1 | 90.7±2.0 | n.a./n.a. | n.a. |
| Citeseer | GCN-MCD | 66.9±3.7/30.4±2.3 | 83.5±1.2 | 73.9±4.4/37.5±1.9 | 91.1±0.8 | 62.5±3.7/51.2±3.2 | 82.1±1.0 |
| | GCN-DE | 62.1±3.4/42.4±3.3 | 80.1±1.7 | 68.6±4.2/51.4±2.9 | 86.4±1.0 | 61.1±3.2/56.2±3.3 | 76.8±1.5 |
| | GCN-MCD+DE | 62.6±2.9/42.8±2.7 | 80.2±1.7 | 68.3±4.3/52.4±2.9 | 86.2±1.1 | 61.2±3.1/56.7±3.0 | 76.8±1.8 |
| | GCN-BNN | 63.6±4.9/58.5±4.3 | 82.5±1.4 | 70.0±3.4/66.6±3.1 | 90.2±0.7 | 64.7±3.4/57.1±2.9 | 81.2±1.3 |
| | GCN-Ens | 67.6±3.0/51.1±4.6 | 83.8±1.1 | 74.1±4.2/64.2±3.2 | 91.2±0.8 | 63.2±3.6/58.1±2.2 | 82.0±1.0 |
| | GPN | 63.6±6.2/64.0±3.3 | 84.9±1.9 | 75.1±4.9/66.6±4.4 | 90.9±0.9 | 57.1±3.3/54.0±3.5 | 83.3±1.9 |
| | SGCN | 66.9±5.5/74.2±5.7 | 83.5±1.9 | 75.5±4.2/84.4±4.6 | 90.6±0.6 | 62.5±3.6/59.1±3.5 | 81.5±1.8 |
| | GCN-EBM | 66.9±3.2/66.3±4.4 | 83.6±1.1 | 73.7±4.3/72.1±4.2 | 91.1±0.9 | 63.4±3.9/60.8±4.4 | 82.0±1.0 |
| | GCN-HEAT | 66.9±3.2/60.6±4.7 | 83.6±1.1 | 73.7±4.3/66.8±4.1 | 91.1±0.9 | 63.4±3.9/60.8±3.4 | 82.0±1.0 |
| | GCNSafe | 66.9±3.2/70.1±4.4 | 83.6±1.1 | 73.7±4.3/72.0±4.5 | 91.1±0.9 | 63.4±3.9/55.9±3.9 | 82.0±1.0 |
| | **GCN-GEBM** | 66.9±3.2/65.8±5.4 | 83.6±1.1 | 73.7±4.3/74.1±3.9 | 91.1±0.9 | 63.4±3.9/59.1±2.5 | 82.0±1.0 |
| PubMed | GCN-MCD | 53.6±2.3/39.2±0.8 | 91.6±1.6 | 28.7±4.0/20.2±0.8 | 83.7±1.5 | n.a./n.a. | n.a. |
| | GCN-DE | 53.7±2.3/44.6±2.4 | 91.2±1.7 | 29.6±2.8/23.9±1.5 | 80.5±2.1 | n.a./n.a. | n.a. |
| | GCN-MCD+DE | 54.3±2.9/44.9±2.7 | 90.8±2.1 | 30.2±3.3/24.5±1.5 | 80.7±1.8 | n.a./n.a. | n.a. |
| | GCN-BNN | 53.8±4.6/51.2±4.8 | 90.7±1.6 | 29.0±4.9/28.4±4.9 | 84.0±1.7 | n.a./n.a. | n.a. |
| | GCN-Ens | 53.5±2.3/42.9±3.1 | 91.7±1.5 | 28.9±4.0/21.2±3.1 | 83.8±1.4 | n.a./n.a. | n.a. |
| | GPN | 52.8±2.5/55.2±6.8 | 92.3±0.8 | 29.5±4.0/39.6±6.6 | 85.6±1.2 | n.a./n.a. | n.a. |
| | SGCN | 54.0±2.3/60.6±3.7 | 92.1±1.6 | 31.3±5.0/40.4±7.2 | 85.1±0.9 | n.a./n.a. | n.a. |
| | GCN-EBM | 53.7±2.4/53.9±2.9 | 91.7±1.5 | 28.7±4.0/28.6±4.0 | 83.7±1.5 | n.a./n.a. | n.a. |
| | GCN-HEAT | 53.7±2.4/50.3±2.7 | 91.7±1.5 | 28.7±4.0/25.9±3.4 | 83.7±1.5 | n.a./n.a. | n.a. |
| | GCNSafe | 53.7±2.4/54.2±3.2 | 91.7±1.5 | 28.7±4.0/34.0±5.6 | 83.7±1.5 | n.a./n.a. | n.a. |
| | **GCN-GEBM** | 53.7±2.4/55.9±3.2 | 91.7±1.5 | 28.7±4.0/30.8±3.5 | 83.7±1.5 | n.a./n.a. | n.a. |
| Amazon Computers | GCN-MCD | 53.2±8.5/28.5±1.3 | 88.0±1.5 | 80.2±3.5/41.4±1.3 | 93.5±2.4 | 58.0±2.3/50.6±1.8 | 81.7±1.1 |
| | GCN-DE | 52.7±9.0/44.0±2.7 | 87.2±1.2 | 81.0±2.9/61.2±1.9 | 93.0±2.7 | 58.4±2.3/52.9±1.6 | 80.7±1.2 |
| | GCN-MCD+DE | 53.6±7.9/43.8±2.2 | 87.1±1.2 | 81.5±3.5/61.8±1.5 | 92.8±2.7 | 58.4±2.1/53.0±1.6 | 80.7±1.3 |
| | GCN-BNN | 44.4±11.7/48.6±11.7 | 85.4±3.2 | 75.4±6.8/75.1±7.7 | 92.3±3.4 | 57.2±1.5/53.0±1.0 | 77.6±2.6 |
| | GCN-Ens | 53.5±9.0/44.3±6.2 | 88.1±1.7 | 79.9±3.8/71.9±5.8 | 93.5±2.6 | 58.2±1.9/49.1±1.1 | 81.7±0.9 |
| | GPN | 68.1±5.3/54.6±4.1 | 86.0±1.4 | 73.6±4.5/69.9±2.1 | 94.4±1.1 | 54.2±1.2/55.1±1.5 | 79.5±2.4 |
| | SGCN | 35.6±15.1/44.3±13.9 | 58.0±29.3 | 67.1±10.7/77.3±9.3 | 92.7±6.5 | 48.0±1.3/49.3±0.5 | 16.1±15.1 |
| | GCN-EBM | 53.0±8.8/50.9±6.4 | 88.0±1.8 | 79.6±3.6/77.3±4.4 | 93.5±2.5 | 58.0±2.0/57.4±2.1 | 81.9±1.0 |
| | GCN-HEAT | 53.0±8.8/45.8±7.0 | 88.0±1.8 | 79.6±3.6/71.2±4.2 | 93.5±2.5 | 58.0±2.0/54.9±2.2 | 81.9±1.0 |
| | GCNSafe | 53.0±8.8/60.9±4.5 | 88.0±1.8 | 79.6±3.6/71.3±2.2 | 93.5±2.5 | 58.0±2.0/53.1±1.5 | 81.9±1.0 |
| | **GCN-GEBM** | 53.0±8.8/66.8±3.5 | 88.0±1.8 | 79.6±3.6/80.4±3.4 | 93.5±2.5 | 58.0±2.0/77.4±2.9 | 81.9±1.0 |
| Amazon Photo | GCN-MCD | 63.9±5.1/41.3±1.4 | 92.0±0.9 | 58.6±5.5/28.2±1.3 | 96.3±0.2 | 59.0±2.9/51.7±1.6 | 90.2±2.0 |
| | GCN-DE | 65.7±5.1/46.4±2.0 | 91.3±1.1 | 57.9±5.0/41.5±3.2 | 95.9±0.4 | 59.4±2.3/55.0±2.3 | 89.5±1.9 |
| | GCN-MCD+DE | 65.4±6.0/46.7±2.2 | 91.5±1.0 | 58.6±5.0/41.5±3.0 | 95.9±0.5 | 59.5±2.5/55.4±2.2 | 89.5±1.7 |
| | GCN-BNN | 68.3±6.7/70.9±10.6 | 93.2±0.8 | 63.8±6.9/73.0±8.8 | 96.5±0.4 | 56.3±2.8/53.6±1.1 | 90.0±2.6 |
| | GCN-Ens | 64.3±5.3/65.6±9.2 | 92.0±0.8 | 58.4±5.4/61.5±9.4 | 96.3±0.2 | 59.2±2.9/53.0±1.2 | 90.4±2.0 |
| | GPN | 67.0±8.3/82.3±3.7 | 89.5±1.2 | 71.1±4.8/48.5±5.9 | 94.3±1.6 | 53.4±2.1/54.4±2.2 | 86.2±2.6 |
| | SGCN | 64.2±5.6/71.3±7.9 | 91.9±0.9 | 50.6±8.2/52.2±9.6 | 96.0±1.0 | 48.6±2.0/49.7±0.9 | 11.4±7.5 |
| | GCN-EBM | 64.1±5.2/64.3±6.3 | 92.0±0.8 | 58.2±5.6/49.1±4.3 | 96.3±0.2 | 58.9±2.7/59.3±2.5 | 90.2±1.9 |
| | GCN-HEAT | 64.1±5.2/61.5±7.8 | 92.0±0.8 | 58.2±5.6/44.5±7.9 | 96.3±0.2 | 58.9±2.7/55.3±2.3 | 90.2±1.9 |
| | GCNSafe | 64.1±5.2/61.4±8.4 | 92.0±0.8 | 58.2±5.6/55.5±7.2 | 96.3±0.2 | 58.9±2.7/55.4±1.6 | 90.2±1.9 |
| | **GCN-GEBM** | 64.1±5.2/77.1±8.1 | 92.0±0.8 | 58.2±5.6/63.2±10.1 | 96.3±0.2 | 58.9±2.7/75.7±4.9 | 90.2±1.9 |
| Coauthor-CS | GCN-MCD | 87.4±1.8/56.0±1.5 | 93.1±0.3 | 70.1±4.8/23.2±1.7 | 94.8±0.4 | 69.5±1.0/53.6±0.9 | 92.1±0.3 |
| | GCN-DE | 87.1±1.7/69.2±1.5 | 92.2±0.5 | 69.9±3.8/34.9±2.4 | 94.0±0.5 | 68.0±1.5/61.8±0.9 | 90.9±0.6 |
| | GCN-MCD+DE | 87.4±1.7/68.9±1.9 | 92.2±0.4 | 69.2±4.6/34.9±2.5 | 93.8±0.6 | 67.7±1.3/61.8±1.2 | 90.8±0.6 |
| | GCN-BNN | 81.8±4.7/80.1±4.7 | 91.9±0.5 | 61.1±6.2/51.6±7.4 | 94.2±0.8 | 67.0±1.6/64.3±1.1 | 91.2±0.5 |
| | GCN-Ens | 87.6±1.9/81.3±2.5 | 93.1±0.3 | 70.3±4.5/53.7±6.6 | 94.8±0.4 | 69.7±0.9/61.0±1.3 | 92.2±0.2 |
| | GPN | 67.6±4.0/90.2±1.7 | 86.7±0.9 | 55.4±8.6/49.8±8.1 | 91.7±1.3 | 55.7±1.2/57.2±1.5 | 85.4±1.7 |
| | SGCN | 47.0±9.5/57.9±8.8 | 29.0±18.1 | 20.5±7.2/12.4±3.3 | 30.3±13.5 | 47.1±0.9/51.2±1.0 | 14.0±8.5 |
| | GCN-EBM | 87.6±2.0/87.8±1.7 | 93.1±0.4 | 69.9±4.6/72.3±4.4 | 94.8±0.4 | 69.4±1.0/69.5±1.3 | 92.0±0.2 |
| | GCN-HEAT | 87.6±2.0/84.6±2.9 | 93.1±0.4 | 69.9±4.6/53.4±10.3 | 94.8±0.4 | 69.4±1.0/68.0±1.6 | 92.0±0.2 |
| | GCNSafe | 87.6±2.0/92.3±1.1 | 93.1±0.4 | 69.9±4.6/80.6±3.7 | 94.8±0.4 | 69.4±1.0/60.9±1.3 | 92.0±0.2 |
| | **GCN-GEBM** | 87.6±2.0/92.8±0.9 | 93.1±0.4 | 69.9±4.6/74.9±7.5 | 94.8±0.4 | 69.4±1.0/94.9±0.7 | 92.0±0.2 |
| Coauthor-Physics | GCN-MCD | 71.8±2.8/18.7±0.5 | 96.6±1.2 | 84.8±4.9/25.0±0.5 | 98.1±0.3 | 59.7±1.5/50.1±0.7 | 92.9±0.7 |
| | GCN-DE | 72.0±2.9/49.1±2.3 | 96.5±1.0 | 83.6±4.8/66.3±2.9 | 97.9±0.4 | 59.0±1.4/56.5±1.3 | 92.3±0.6 |
| | GCN-MCD+DE | 71.0±3.5/48.3±2.0 | 96.5±1.0 | 84.0±4.6/66.8±2.5 | 97.9±0.3 | 59.0±1.5/56.3±1.5 | 92.3±0.7 |
| | GCN-BNN | 64.1±5.4/59.7±5.3 | 96.6±1.0 | 73.6±8.3/76.0±8.0 | 97.9±0.3 | 58.8±1.8/57.5±1.7 | 92.4±0.9 |
| | GCN-Ens | 71.9±2.7/55.1±6.7 | 96.7±1.2 | 84.9±4.9/72.9±5.5 | 98.1±0.3 | 59.7±1.4/55.4±1.3 | 92.9±0.7 |
| | GPN | 61.3±5.2/74.9±4.1 | 96.8±0.5 | 57.8±5.8/87.1±2.6 | 97.7±0.3 | 55.1±1.3/55.7±1.5 | 92.3±0.7 |
| | SGCN | 71.3±3.5/73.7±8.2 | 96.5±1.0 | 86.1±3.3/88.3±3.5 | 98.1±0.4 | 60.2±1.9/60.6±2.2 | 92.4±0.9 |
| | GCN-EBM | 71.9±2.6/75.3±3.5 | 96.7±1.1 | 84.8±4.9/88.0±4.0 | 98.1±0.3 | 59.8±1.5/59.4±1.7 | 92.8±0.7 |
| | GCN-HEAT | 71.9±2.6/58.2±4.8 | 96.7±1.1 | 84.8±4.9/73.9±6.6 | 98.1±0.3 | 59.8±1.5/57.7±2.0 | 92.8±0.7 |
| | GCNSafe | 71.9±2.6/83.5±2.2 | 96.7±1.1 | 84.8±4.9/92.1±3.3 | 98.1±0.3 | 59.8±1.5/55.3±1.4 | 92.8±0.7 |
| | **GCN-GEBM** | 71.9±2.6/80.6±3.7 | 96.7±1.1 | 84.8±4.9/91.5±4.3 | 98.1±0.3 | 59.8±1.5/78.6±3.5 | 92.8±0.7 |

| Dataset | Model | Ber(0.5) AUC-PR↑ (Alea. / Epi.) | Ber(0.5) Acc.↑ | $\mathcal{N}(0,1)$ (far) AUC-PR↑ (Alea. / Epi.) | $\mathcal{N}(0,1)$ (far) Acc.↑ | Page Rank AUC-PR↑ (Alea. / Epi.) | Page Rank Acc.↑ | Homophily AUC-PR↑ (Alea. / Epi.) | Homophily Acc.↑ |
|---|---|---|---|---|---|---|---|---|---|
| CoraML | GCN-MCD | 56.5±5.1/50.3±2.9 | 82.0±0.8 | 37.4±1.7/48.0±2.7 | 82.0±0.8 | 70.0±1.1/56.5±2.7 | 82.6±1.5 | 63.7±1.8/48.3±1.8 | 93.9±0.8 |
| | GCN-DE | 48.3±3.5/71.0±3.3 | 78.4±1.9 | 35.6±1.7/59.3±3.8 | 78.4±1.9 | 64.1±1.8/68.1±1.8 | 79.0±2.3 | 63.4±3.5/50.5±1.7 | 90.6±1.8 |
| | GCN-MCD+DE | 48.7±3.7/71.2±3.6 | 78.1±2.2 | 35.8±1.4/61.1±2.8 | 78.1±2.2 | 64.5±1.6/67.5±1.9 | 79.5±1.7 | 63.5±2.6/50.1±1.4 | 90.8±1.7 |
| | GCN-BNN | 47.9±4.4/65.8±6.0 | 80.6±2.0 | 37.5±1.7/55.2±3.6 | 80.6±2.0 | 67.6±1.6/53.8±3.0 | 80.4±2.3 | 63.3±2.0/64.5±2.4 | 93.1±1.5 |
| | GCN-Ens | 56.0±4.4/62.3±3.7 | 81.8±0.8 | 37.5±1.4/54.9±3.5 | 81.8±0.8 | 70.1±1.0/55.3±2.3 | 82.7±1.5 | 63.6±1.3/64.0±2.9 | 94.0±0.5 |
| | GPN | 49.8±3.5/51.4±4.4 | 78.8±2.1 | 50.7±2.2/56.0±4.2 | 78.8±2.1 | 55.6±1.6/60.5±1.5 | 78.4±1.8 | 67.9±3.8/49.1±1.6 | 91.1±1.8 |
| | SGCN | 45.3±10.3/62.0±4.2 | 51.7±30.2 | 36.2±1.1/44.3±8.6 | 51.7±30.2 | 69.3±1.6/62.6±2.3 | 83.3±1.3 | 64.1±1.8/65.8±1.8 | 94.2±1.0 |
| | GCN-EBM | 57.1±5.5/56.3±6.2 | 81.9±0.8 | 37.5±1.7/35.0±1.4 | 81.9±0.8 | 70.0±1.0/70.2±2.9 | 82.5±1.6 | 63.4±1.5/62.5±2.1 | 94.1±0.6 |
| | GCN-HEAT | 57.1±5.5/59.9±5.7 | 81.9±0.8 | 37.5±1.7/73.6±3.0 | 81.9±0.8 | 70.0±1.0/67.9±2.5 | 82.5±1.6 | 63.4±1.5/63.8±2.7 | 94.1±0.6 |
| | GCNSafe | 57.1±5.5/52.9±3.9 | 81.9±0.8 | 37.5±1.7/38.5±1.4 | 81.9±0.8 | 70.0±1.0/53.0±0.7 | 82.5±1.6 | 63.4±1.5/60.9±2.0 | 94.1±0.6 |
| | **GCN-GEBM** | 57.1±5.5/91.6±5.2 | 81.9±0.8 | 37.5±1.7/91.7±1.8 | 81.9±0.8 | 70.0±1.0/53.3±0.7 | 82.5±1.6 | 63.4±1.5/64.3±1.4 | 94.1±0.6 |
| CoraML-LLM | GCN-MCD | 70.2±3.5/53.1±2.9 | 82.6±1.5 | 42.9±2.9/49.4±3.0 | 82.6±1.5 | 70.7±1.4/55.6±2.1 | 84.9±2.1 | 64.8±1.4/47.8±1.9 | 94.1±1.1 |
| | GCN-DE | 75.8±2.4/40.5±1.8 | 79.9±1.8 | 42.2±2.2/65.0±2.9 | 79.9±1.8 | 64.7±1.6/68.9±1.4 | 82.2±2.7 | 64.0±2.0/53.5±1.8 | 91.7±1.4 |
| | GCN-MCD+DE | 75.9±3.3/40.7±1.6 | 80.3±1.8 | 41.1±2.2/64.2±2.3 | 80.3±1.8 | 64.6±1.3/69.4±1.4 | 82.7±2.6 | 63.6±2.3/53.8±1.8 | 91.4±1.5 |
| | GCN-BNN | 72.3±2.4/43.2±2.4 | 82.8±1.6 | 41.9±1.7/64.6±2.2 | 82.8±1.6 | 68.7±1.5/51.1±1.8 | 84.0±2.2 | 67.2±1.6/68.8±2.9 | 94.0±1.3 |
| | GCN-Ens | 70.8±3.8/52.5±3.0 | 82.6±1.5 | 41.1±2.2/63.7±3.2 | 82.6±1.5 | 70.8±1.4/58.3±1.8 | 85.0±2.1 | 64.9±1.3/63.4±2.5 | 94.0±1.1 |
| | GPN | 53.2±2.2/54.1±2.4 | 79.7±1.9 | 52.3±2.3/55.5±2.7 | 79.7±1.8 | 55.2±1.1/61.0±0.8 | 80.9±2.2 | 71.0±2.2/51.3±1.7 | 92.5±1.0 |
| | SGCN | 64.4±3.3/53.0±3.8 | 68.7±21.0 | 41.5±1.9/47.2±6.3 | 68.7±21.0 | 68.5±1.4/64.9±2.4 | 82.0±2.7 | 63.4±1.4/64.4±3.2 | 93.1±1.1 |
| | GCN-EBM | 70.6±3.3/66.0±4.4 | 82.6±1.6 | 42.3±2.2/41.0±2.8 | 82.6±1.6 | 70.6±1.4/69.6±3.5 | 85.0±2.2 | 65.0±1.4/64.7±1.6 | 93.8±1.0 |
| | GCN-HEAT | 70.6±3.3/67.6±5.4 | 82.6±1.6 | 42.3±2.2/66.8±3.5 | 82.6±1.6 | 70.6±1.4/64.0±3.5 | 85.0±2.2 | 65.0±1.4/64.8±2.4 | 93.8±1.0 |
| | GCNSafe | 70.6±3.3/57.4±3.0 | 82.6±1.6 | 42.3±2.2/45.2±3.3 | 82.6±1.6 | 70.6±1.4/54.9±0.9 | 85.0±2.2 | 65.0±1.4/61.9±1.8 | 93.8±1.0 |
| | **GCN-GEBM** | 70.6±3.3/72.4±3.3 | 82.6±1.6 | 42.3±2.2/88.4±2.5 | 82.6±1.6 | 70.6±1.4/52.5±0.4 | 85.0±2.2 | 65.0±1.4/65.2±1.9 | 93.8±1.0 |
| Citeseer | GCN-MCD | 58.2±4.1/51.3±3.0 | 82.1±1.0 | 38.4±2.2/49.0±2.7 | 82.1±1.0 | 60.3±0.9/50.7±2.0 | 82.9±1.1 | 51.5±0.7/50.6±1.5 | 86.3±1.5 |
| | GCN-DE | 46.3±4.1/80.8±2.6 | 76.8±1.5 | 35.8±2.4/64.9±2.2 | 76.8±1.5 | 56.2±1.1/55.9±1.9 | 79.0±1.3 | 50.6±1.1/50.5±1.0 | 81.6±1.6 |
| | GCN-MCD+DE | 47.1±2.7/81.4±1.7 | 76.8±1.8 | 35.8±2.4/64.6±3.4 | 76.8±1.8 | 56.2±1.3/56.0±1.6 | 79.2±1.0 | 50.7±0.8/50.3±1.2 | 81.7±1.5 |
| | GCN-BNN | 50.2±3.5/72.4±4.2 | 81.2±1.3 | 37.2±2.2/59.9±2.1 | 81.2±1.3 | 58.6±0.8/48.6±0.7 | 81.3±1.1 | 51.9±0.8/53.1±0.8 | 86.6±1.1 |
| | GCN-Ens | 57.0±2.7/79.8±2.9 | 82.0±1.0 | 38.0±2.2/58.1±3.0 | 82.0±1.0 | 60.4±1.0/45.5±0.8 | 82.6±0.8 | 51.5±0.7/54.9±1.5 | 86.4±1.6 |
| | GPN | 52.3±3.3/57.0±6.8 | 83.3±1.9 | 56.7±2.4/66.5±3.1 | 83.3±1.9 | 45.7±0.7/49.1±2.1 | 82.1±1.5 | 53.6±0.6/50.5±0.9 | 88.0±1.0 |
| | SGCN | 60.5±5.5/64.2±7.6 | 81.5±1.8 | 36.8±2.2/35.8±2.5 | 81.5±1.8 | 59.1±1.1/55.6±2.9 | 82.5±1.0 | 51.5±0.9/52.4±1.1 | 85.9±1.4 |
| | GCN-EBM | 57.7±5.5/50.0±5.0 | 82.0±1.0 | 38.4±2.2/35.3±2.5 | 82.0±1.0 | 60.2±1.0/60.2±1.8 | 82.4±1.0 | 51.4±0.7/51.3±0.7 | 86.4±1.5 |
| | GCN-HEAT | 57.7±5.5/75.8±3.0 | 82.0±1.0 | 38.4±2.2/77.8±2.6 | 82.0±1.0 | 60.2±1.0/56.3±1.2 | 82.4±1.0 | 51.4±0.7/53.4±1.1 | 86.4±1.5 |
| | GCNSafe | 57.7±5.5/49.3±3.9 | 82.0±1.0 | 38.4±2.2/38.4±2.4 | 82.0±1.0 | 60.2±1.0/45.3±0.4 | 82.4±1.0 | 51.4±0.7/52.4±0.5 | 86.4±1.5 |
| | **GCN-GEBM** | 57.7±5.5/98.6±0.8 | 82.0±1.0 | 38.4±2.2/83.3±2.5 | 82.0±1.0 | 60.2±1.0/47.6±1.0 | 82.4±1.0 | 51.4±0.7/51.4±0.3 | 86.4±1.5 |
| PubMed | GCN-MCD | 62.8±7.0/50.5±1.0 | 78.1±1.0 | 38.0±0.4/49.7±1.0 | 78.1±1.0 | 62.1±0.6/49.9±1.0 | 77.9±2.6 | 56.0±0.6/49.4±0.7 | 88.9±2.8 |
| | GCN-DE | 57.4±5.5/72.0±2.8 | 74.9±2.1 | 36.6±0.0/56.0±1.5 | 74.9±2.1 | 54.9±0.8/66.1±1.1 | 76.0±2.6 | 55.4±1.1/48.6±0.7 | 84.8±2.3 |
| | GCN-MCD+DE | 56.8±5.5/72.2±2.2 | 75.4±1.5 | 36.6±0.0/56.0±1.3 | 75.4±1.5 | 55.0±0.6/65.7±0.9 | 76.3±1.9 | 55.3±0.9/48.8±0.7 | 84.9±1.8 |
| | GCN-BNN | 56.1±4.7/62.4±4.6 | 78.2±1.6 | 38.6±0.0/56.1±3.0 | 78.2±1.6 | 59.9±1.0/45.7±2.2 | 76.1±4.2 | 56.1±1.3/60.2±1.5 | 89.3±1.9 |
| | GCN-Ens | 62.2±6.9/69.0±2.4 | 78.1±0.9 | 37.9±0.0/55.5±2.3 | 78.1±0.9 | 62.2±0.6/46.1±2.9 | 77.9±2.6 | 56.0±0.6/59.3±2.1 | 88.9±2.8 |
| | GPN | 53.7±4.3/53.7±3.8 | 79.8±1.1 | 52.5±1.4/65.9±5.0 | 79.8±1.1 | 47.5±0.6/65.5±3.2 | 80.2±1.4 | 60.8±1.4/47.5±0.8 | 90.3±1.7 |
| | SGCN | 64.5±4.4/57.2±6.9 | 80.1±0.9 | 38.5±0.0/38.6±0.9 | 80.1±0.9 | 61.8±0.9/70.1±2.8 | 79.3±2.8 | 56.9±0.8/55.9±1.9 | 90.7±1.8 |
| | GCN-EBM | 62.6±6.6/53.1±5.2 | 78.1±1.0 | 38.0±0.0/38.0±0.4 | 78.1±1.0 | 62.2±0.6/64.4±4.1 | 78.0±2.6 | 56.1±0.6/55.1±1.3 | 88.8±2.7 |
| | GCN-HEAT | 62.6±6.6/54.7±6.2 | 78.1±1.0 | 38.0±0.0/77.1±3.5 | 78.1±1.0 | 62.2±0.6/53.4±5.5 | 78.0±2.6 | 56.1±0.6/60.0±2.9 | 88.8±2.7 |
| | GCNSafe | 62.6±6.6/51.4±1.8 | 78.1±1.0 | 38.0±0.0/41.5±0.6 | 78.1±1.0 | 62.2±0.6/50.1±0.7 | 78.0±2.6 | 56.1±0.6/57.6±0.7 | 88.8±2.7 |
| | **GCN-GEBM** | 62.6±6.6/83.3±6.2 | 78.1±1.0 | 38.0±0.0/77.3±5.3 | 78.1±1.0 | 62.2±0.6/48.4±0.6 | 78.0±2.6 | 56.1±0.6/54.3±1.5 | 88.8±2.7 |
| Amazon Computers | GCN-MCD | 54.0±3.9/50.4±1.6 | 81.7±1.1 | 44.1±1.4/48.2±1.0 | 81.7±1.1 | 81.6±1.2/54.7±1.3 | 82.7±1.1 | 60.5±0.5/51.6±0.9 | 92.3±0.7 |
| | GCN-DE | 54.5±2.2/52.0±1.7 | 80.7±1.2 | 44.3±1.3/55.6±1.1 | 80.7±1.2 | 78.7±1.1/69.3±0.5 | 82.0±1.0 | 62.0±0.5/50.6±0.5 | 90.9±0.8 |
| | GCN-MCD+DE | 54.7±2.2/51.9±1.7 | 80.7±1.3 | 44.7±1.3/55.4±1.4 | 80.7±1.3 | 78.9±0.9/69.3±0.5 | 81.7±1.5 | 62.2±1.0/50.7±0.4 | 91.1±0.7 |
| | GCN-BNN | 51.8±2.2/51.0±1.6 | 77.6±2.6 | 43.6±1.1/57.9±1.7 | 77.6±2.6 | 77.9±1.1/45.1±2.5 | 80.7±2.4 | 60.2±1.4/64.0±2.0 | 90.9±1.6 |
| | GCN-Ens | 54.1±3.3/49.9±1.4 | 81.7±0.9 | 43.5±1.1/56.6±0.6 | 81.7±0.9 | 81.8±1.0/54.7±2.4 | 82.6±1.0 | 60.5±0.5/57.4±1.3 | 92.4±0.6 |
| | GPN | 54.3±1.3/56.3±1.8 | 79.5±2.4 | 55.7±1.7/57.9±1.7 | 79.5±2.4 | 50.7±0.9/71.4±1.6 | 79.3±2.0 | 66.1±3.3/48.9±0.7 | 86.1±14.4 |
| | SGCN | 47.3±1.1/49.9±0.5 | 16.1±15.1 | 48.0±3.3/47.9±1.7 | 16.1±15.1 | 93.7±0.9/35.8±16.2 | 14.9±13.4 | 45.4±0.3/58.8±2.1 | 16.7±10.6 |
| | GCN-EBM | 54.2±3.5/53.3±2.9 | 81.9±1.0 | 44.4±1.3/43.7±1.2 | 81.9±1.0 | 81.8±1.1/84.4±2.5 | 82.6±1.2 | 60.6±0.5/59.4±0.9 | 92.4±0.7 |
| | GCN-HEAT | 54.2±3.5/52.3±3.1 | 81.9±1.0 | 44.4±1.3/60.9±1.4 | 81.9±1.0 | 81.8±1.1/72.5±6.1 | 82.6±1.2 | 60.6±0.5/62.0±1.4 | 92.4±0.7 |
| | GCNSafe | 54.2±3.5/50.9±1.8 | 81.9±1.0 | 44.4±1.3/44.1±1.1 | 81.9±1.0 | 81.8±1.1/54.6±1.0 | 82.6±1.2 | 60.6±0.5/52.3±0.5 | 92.4±0.7 |
| | **GCN-GEBM** | 54.2±3.5/64.4±2.9 | 81.9±1.0 | 44.4±1.3/94.7±0.9 | 81.9±1.0 | 81.8±1.1/49.7±1.1 | 82.6±1.2 | 60.6±0.5/59.5±1.5 | 92.4±0.7 |
| Amazon Photo | GCN-MCD | 52.9±3.5/50.5±1.8 | 90.2±2.0 | 45.0±0.8/48.9±1.2 | 90.2±2.0 | 83.7±0.8/53.6±1.8 | 91.8±0.6 | 64.0±1.4/50.7±1.2 | 97.6±0.7 |
| | GCN-DE | 53.3±2.2/52.2±2.9 | 89.5±1.9 | 45.2±1.4/54.0±1.9 | 89.5±1.9 | 81.0±0.9/75.3±0.5 | 91.2±0.7 | 63.8±1.4/51.1±0.7 | 96.9±0.8 |
| | GCN-MCD+DE | 53.4±2.2/52.8±3.2 | 89.5±1.7 | 44.8±0.9/54.4±1.7 | 89.5±1.7 | 80.9±0.9/75.3±0.7 | 91.2±0.7 | 63.9±1.4/51.1±0.8 | 96.9±0.8 |
| | GCN-BNN | 51.7±4.4/49.5±2.5 | 90.0±2.6 | 45.2±1.4/58.9±0.8 | 90.0±2.6 | 76.1±2.2/55.4±3.7 | 90.7±1.4 | 68.0±2.5/68.7±2.8 | 97.9±0.9 |
| | GCN-Ens | 52.8±2.2/49.9±1.1 | 90.4±2.0 | 45.2±2.2/56.1±1.6 | 90.4±2.0 | 83.8±0.9/65.1±2.9 | 91.9±0.5 | 64.1±1.3/57.6±2.0 | 97.7±0.6 |
| | GPN | 53.5±2.2/54.6±2.4 | 86.2±2.6 | 54.5±2.2/55.8±2.4 | 86.2±2.6 | 51.8±1.0/70.8±2.0 | 88.8±1.9 | 70.5±2.3/48.2±1.6 | 96.4±0.9 |
| | SGCN | 47.7±1.1/50.4±1.0 | 11.4±7.5 | 48.6±2.2/49.1±2.2 | 11.4±7.5 | 84.9±5.5/66.3±22.2 | 64.8±36.0 | 45.8±0.9/56.7±3.1 | 12.2±7.0 |
| | GCN-EBM | 52.8±2.2/52.0±2.1 | 90.2±1.9 | 44.8±1.8/44.3±2.2 | 90.2±1.9 | 83.8±1.0/89.5±1.4 | 91.9±0.6 | 64.0±1.4/58.9±1.9 | 97.7±0.6 |
| | GCN-HEAT | 52.8±2.2/50.2±2.2 | 90.2±1.9 | 44.8±1.8/60.2±1.8 | 90.2±1.9 | 83.8±1.0/75.6±6.0 | 91.9±0.6 | 64.0±1.4/59.9±2.5 | 97.7±0.6 |
| | GCNSafe | 52.8±2.2/51.9±1.9 | 90.2±1.9 | 44.8±1.8/45.8±1.7 | 90.2±1.9 | 83.8±1.0/59.9±0.6 | 91.9±0.6 | 64.0±1.4/52.6±2.1 | 97.7±0.6 |
| | **GCN-GEBM** | 52.8±2.2/51.9±3.5 | 90.2±1.9 | 44.8±1.8/94.4±1.7 | 90.2±1.9 | 83.8±1.0/55.3±0.9 | 91.9±0.6 | 64.0±1.4/58.6±2.0 | 97.7±0.6 |
| Coauthor-CS | GCN-MCD | 70.8±1.5/53.2±0.9 | 92.1±0.3 | 41.0±0.7/48.7±0.9 | 92.1±0.3 | 73.4±0.7/54.1±1.0 | 92.6±0.5 | 65.1±0.3/50.1±0.9 | 98.8±0.4 |
| | GCN-DE | 62.6±2.0/69.5±1.0 | 90.9±0.6 | 40.6±0.7/60.5±0.8 | 90.9±0.6 | 67.1±0.6/75.2±0.8 | 90.9±0.6 | 63.2±0.6/54.7±0.8 | 98.2±0.3 |
| | GCN-MCD+DE | 62.9±1.7/69.6±0.9 | 90.8±0.6 | 40.6±0.7/60.6±1.1 | 90.8±0.6 | 67.1±0.6/75.2±0.7 | 90.9±0.7 | 63.3±0.6/54.8±0.7 | 98.2±0.3 |
| | GCN-BNN | 51.1±3.5/66.7±4.5 | 91.2±0.5 | 37.7±0.7/54.2±1.4 | 91.2±0.5 | 66.1±0.7/53.7±0.8 | 91.7±0.8 | 70.0±0.9/73.0±1.1 | 98.8±0.5 |
| | GCN-Ens | 70.9±1.7/67.8±1.4 | 92.2±0.2 | 39.7±0.6/61.6±1.0 | 92.2±0.2 | 73.4±0.6/66.7±1.4 | 92.7±0.5 | 65.1±0.3/68.8±0.8 | 98.9±0.4 |
| | GPN | 54.3±0.8/58.7±2.1 | 85.4±1.7 | 54.3±0.8/58.8±2.1 | 85.4±1.7 | 49.6±0.6/76.5±2.0 | 86.7±1.5 | 66.7±1.8/45.3±0.6 | 94.8±1.2 |
| | SGCN | 39.3±0.6/61.0±1.4 | 14.0±8.5 | 40.3±1.1/52.9±1.6 | 14.0±8.5 | 81.1±4.7/34.0±0.5 | 15.1±9.2 | 40.5±1.1/62.2±0.9 | 13.1±9.3 |
| | GCN-EBM | 70.3±1.1/72.2±1.9 | 92.0±0.2 | 40.9±0.6/38.0±0.6 | 92.0±0.2 | 73.4±0.7/77.5±1.6 | 92.6±0.5 | 65.1±0.3/60.5±0.4 | 98.9±0.4 |
| | GCN-HEAT | 70.3±1.1/67.1±1.5 | 92.0±0.2 | 40.9±0.6/69.5±1.2 | 92.0±0.2 | 73.4±0.7/65.1±4.8 | 92.6±0.5 | 65.1±0.3/65.7±1.2 | 98.9±0.4 |
| | GCNSafe | 70.3±1.1/62.9±1.4 | 92.0±0.2 | 40.9±0.6/41.9±0.7 | 92.0±0.2 | 73.4±0.7/61.5±0.8 | 92.6±0.5 | 65.1±0.3/58.1±0.5 | 98.9±0.4 |
| | **GCN-GEBM** | 70.3±1.1/99.2±0.6 | 92.0±0.2 | 40.9±0.6/96.4±1.4 | 92.0±0.2 | 73.4±0.7/54.9±0.6 | 92.6±0.5 | 65.1±0.3/64.7±1.3 | 98.9±0.4 |
| Coauthor-Physics | GCN-MCD | 61.5±1.1/50.4±0.6 | 92.9±0.7 | 44.8±0.9/49.4±0.9 | 92.9±0.7 | 74.2±0.8/50.0±0.6 | 94.5±0.8 | 60.2±0.4/49.9±0.5 | 98.8±0.2 |
| | GCN-DE | 61.1±1.3/60.9±1.2 | 92.3±0.6 | 45.0±0.4/55.4±0.8 | 92.3±0.6 | 69.5±0.8/75.3±0.6 | 94.1±0.7 | 59.5±0.5/55.0±0.3 | 98.4±0.3 |
| | GCN-MCD+DE | 60.7±1.1/60.9±1.3 | 92.3±0.7 | 44.6±0.5/53.1±0.8 | 92.3±0.7 | 69.5±0.8/75.2±0.6 | 94.2±0.7 | 59.5±0.5/55.1±0.5 | 98.4±0.3 |
| | GCN-BNN | 48.5±3.5/58.2±4.7 | 92.4±0.9 | 40.2±0.6/54.4±1.4 | 92.4±0.9 | 69.6±0.6/56.8±2.0 | 94.2±1.0 | 63.3±0.7/66.3±0.9 | 98.9±0.3 |
| | GCN-Ens | 61.8±1.1/58.3±1.1 | 92.9±0.7 | 44.1±0.6/54.4±1.2 | 92.9±0.7 | 74.2±0.8/68.0±2.7 | 94.5±0.8 | 60.2±0.4/65.6±0.8 | 98.7±0.3 |
| | GPN | 53.5±1.1/57.5±2.1 | 92.3±0.7 | 53.3±1.1/57.7±2.0 | 92.3±0.7 | 51.2±0.5/77.4±2.0 | 93.3±0.8 | 69.7±1.5/49.0±0.6 | 98.1±0.4 |
| | SGCN | 61.2±1.1/60.0±2.8 | 92.4±0.9 | 44.8±0.4/44.2±0.8 | 92.4±0.9 | 73.4±1.4/76.2±3.7 | 94.0±0.7 | 59.8±0.5/58.3±2.2 | 98.6±0.4 |
| | GCN-EBM | 61.7±1.1/61.1±1.9 | 92.8±0.7 | 44.7±0.4/43.3±0.6 | 92.8±0.7 | 74.0±1.0/78.4±1.4 | 94.5±0.7 | 60.2±0.4/57.5±0.6 | 98.7±0.2 |
| | GCN-HEAT | 61.7±1.1/60.4±2.0 | 92.8±0.7 | 44.7±0.4/60.2±1.0 | 92.8±0.7 | 74.0±1.0/69.5±2.0 | 94.5±0.7 | 60.2±0.4/59.3±0.6 | 98.7±0.2 |
| | GCNSafe | 61.7±1.1/56.3±1.2 | 92.8±0.7 | 44.7±0.4/56.4±0.7 | 92.8±0.7 | 74.0±1.0/63.4±0.8 | 94.5±0.7 | 60.2±0.4/56.7±0.5 | 98.7±0.2 |
| | **GCN-GEBM** | 61.7±1.1/90.9±4.1 | 92.8±0.7 | 44.7±0.4/84.4±2.7 | 92.8±0.7 | 74.0±1.0/56.7±0.4 | 94.5±0.7 | 60.2±0.4/61.7±0.8 | 98.7±0.2 |

Table 12: Out-of-Distribution detection AUC-PR (↑) using aleatoric or epistemic uncertainty (best and runner-up) in a transductive setting.

We apply GEBM to an APPNP model and expose it to increasingly severe feature perturbations. Even though APPNP uses the same number of diffusion steps as GEBM ($t = 10$), it suffers from deteriorating accuracy even when a small portion of node features is perturbed. Since logits and energy are unbounded and likely to diverge under strong enough distribution shifts (see Proposition 4.1), a fixed number of diffusion steps is insufficient to recover from perturbations. At the same time, the regularized energy of GEBM converges to zero and, therefore, a fixed number of diffusion steps can effectively maintain high robustness akin to evidential methods like GPN.

## C.10 Partially Perturbing Node Features

The feature perturbation-based distribution shifts discussed in Section 5.1 replace node features with noise. We additionally study a feature-level distribution shift that partially corrupts the features of an instance. To that end, we vary the fraction $p$ with which an individual node feature is perturbed and again measure how well uncertainty estimators can detect this distribution shift. In particular, we replace $d \cdot p$ uniformly and independently selected node feature $\boldsymbol{x}_i$ with Gaussian noise and keep $d - d \cdot p$ features as-is.

| Model | CoraML | CoraML LLM | Citeseer | PubMed | Amazon Computers | Amazon Photo | Coauthor CS | Coauthor Physics |
|---|---|---|---|---|---|---|---|---|
| GCN-MCD | 9.9/18.9 | 9.4/19.0 | 9.1/18.5 | 9.5/19.1 | 8.9/18.2 | 10.1/19.4 | 10.0/18.9 | 10.5/20.3 |
| GCN-DE | 6.3/13.9 | 7.4/16.1 | 7.2/15.6 | 6.8/12.9 | 7.6/16.7 | 7.3/16.4 | 6.7/13.3 | 7.1/14.4 |
| GCN-MCD+DE | 6.7/14.5 | 7.5/16.2 | 6.9/15.3 | 6.1/12.3 | 7.6/16.7 | 6.2/15.3 | 6.0/12.7 | 7.2/14.7 |
| GCN-BNN | 5.7/12.1 | 6.7/14.4 | 6.2/12.8 | 5.6/12.8 | 5.3/12.6 | 4.9/**8.9** | 6.0/12.6 | 5.7/12.9 |
| GCN-Ens | 7.2/14.2 | 6.8/16.1 | 6.3/12.9 | 7.6/13.9 | 7.2/16.1 | 5.9/12.3 | 4.5/11.0 | 6.4/13.8 |
| GPN | 7.0/15.1 | 6.2/12.9 | 7.0/15.4 | 5.1/**8.4** | 5.1/**11.2** | 5.4/11.6 | 6.6/12.0 | 5.9/9.8 |
| SGCN | **3.9**/**6.9** | **4.2**/**8.8** | 4.2/**8.4** | **3.5**/**6.3** | 7.6/16.5 | 6.3/13.7 | 9.3/18.4 | 4.7/**8.5** |
| GCN-EBM | 4.8/9.3 | 4.4/11.2 | 4.8/13.1 | 6.4/13.3 | **3.8**/11.3 | **4.6**/11.3 | **4.0**/**7.1** | **4.1**/8.6 |
| GCN-HEAT | 4.7/10.2 | 4.4/12.9 | **3.5**/9.8 | 5.2/12.7 | 3.9/11.3 | 4.9/14.1 | 4.4/10.7 | 5.2/14.0 |
| GCNSafe | 6.7/12.1 | 5.3/10.8 | 6.7/12.5 | 5.8/9.5 | 6.4/14.7 | 6.5/15.6 | 5.7/10.9 | 6.1/11.8 |
| **GCN-GEBM** | **3.2** / **5.2** | **3.6** / **6.5** | **4.2** / **9.3** | **4.4** / 8.5 | **2.7** / **6.7** | **3.9** / **8.6** | **2.8** / **5.8** | **3.2** / **5.2** |

Table 13: Average o.o.d. detection rank (AUC-PR) (↓) of epistemic uncertainty versus other epistemic measures / all uncertainty measures over all distribution shifts in a transductive setting (best and runner-up).

| Model | GCN MCD | GCN DE | GCN MCD+DE | GCN BNN | GCN Ens | GPN | SGCN | GCN EBM | GCN HEAT | GCNSafe | GCN GEBM |
|---|---|---|---|---|---|---|---|---|---|---|---|
| Global Rank ↓ | 19.6 | 15.3 | 15.2 | 10.7 | 12.6 | 13.0 | 13.9 | **9.4** | 10.3 | 10.6 | **5.0** |

Table 14: Global o.o.d. detection rank of each model, averaged over all datasets and distribution shifts in an inductive setting based on AUC-ROC scores (best and runner-up).

We compare the AUC-ROC of different uncertainty estimators while increasing the fraction of perturbed features for each o.o.d. node in Figure 7. While both the softmax-level uncertainty of the vanilla GCN and the EBM again become overconfident, GEBM can reliably identify this distribution shift. This coincides with observations made for feature perturbations that affect all node features simultaneously and justifies focusing on that distribution shift for the bulk of our study.

# D Ablations

## D.1 Energy at Different Scales

We report AUC-ROC and AUC-PR for o.o.d.-detection ablating GEBM in an inductive and transductive setting with standard deviations in Tables 26 to 29. Additionally, we rank all variants of GEBM (i.e. energies at different structural scales) similar to the o.o.d. detection experiments in Section 5.2. We observe our proposed GEBM to consistently be the most effective over different shifts simultaneously. We can also confirm the effectiveness of the scale-specific energies at detecting distribution shifts that should be covered from their definition. Both the best and second-best ranking methods are GEBM and a variant that uses unregularized energy: This shows the efficacy of a scale-aware EBM for graph problems.

## D.2 Diffusion Process

We also ablate the diffusion operator $P_{\boldsymbol{A}}$ and its hyperparameters, i.e. the type of diffusion process (see Appendix C.6), the number of diffusion steps $t$ as well the teleport probability $\alpha$, in Figure 8. We evaluate the performance of GEBM on a representative of each distribution shift family. As expected, low $t$ and high $\alpha$ aid node-level anomaly detection while the opposite holds for cluster and local shifts. Label-Propagation achieves satisfactory performance over the entire range of diffusion hyperparameters which justifes using it in our experiments. In particular, we did not tune any hyperparameters for good o.o.d.-detection. As stated in Appendix C.6, it does not bias the energy toward high-degree nodes which explains its advantages over the other diffusion types. It performs well over a broad range of hyperparameters making GEBM less sensitive to those.

| Dataset | LoC *(last)* Abs. / Rel. | LoC *(hetero.)* Abs. / Rel. | Ber($\hat{p}$) *(near)* Abs. / Rel. | Ber(0.5) Abs. / Rel. | $\mathcal{N}(0,1)$ *(far)* Abs. / Rel. | Page Rank Abs. / Rel. | Homophily Abs. / Rel. | (Weighted) Avg. Abs. / Rel. |
|---|---|---|---|---|---|---|---|---|
| CoraML | +1.7 / +1.9% | +1.0 / +1.2% | +10.4 / +15.5% | +35.9 / +61.3% | +60.6 / +235.7% | −16.5 / −22.3% | +5.5 / +7.7% | +10.5 / +32.8% |
| CoraML-LLM | +1.3 / +1.5% | +2.2 / +2.5% | n.a. | +12.2 / +17.5% | +52.3 / +158.2% | −20.8 / −28.6% | +5.3 / +7.6% | +5.8 / +18.8% |
| Citeseer | +1.5 / +1.7% | +0.8 / +0.9% | +1.2 / +1.9% | +52.5 / +119.9% | +48.6 / +207.1% | −18.1 / −27.3% | +1.7 / +3.3% | +9.0 / +33.0% |
| PubMed | +5.1 / +7.8% | −0.6 / −0.9% | n.a. | +35.4 / +72.7% | +48.2 / +175.3% | −23.2 / −31.9% | +6.7 / +12.9% | +8.2 / +28.7% |
| Amazon Computers | +5.9 / +7.8% | +2.4 / +2.8% | +28.7 / +50.7% | +20.5 / +37.3% | +54.0 / +134.9% | −32.7 / −38.3% | +2.9 / +4.7% | +7.9 / +20.9% |
| Amazon Photo | +8.4 / +11.2% | +9.4 / +12.0% | +26.9 / +46.6% | +12.2 / +23.4% | +49.8 / +116.8% | −32.3 / −35.7% | +8.4 / +14.0% | +8.9 / +21.0% |
| Coauthor-CS | +2.9 / +3.2% | +1.9 / +2.0% | +26.0 / +37.3% | +28.0 / +39.1% | +61.2 / +212.8% | −25.4 / −31.0% | +9.0 / +14.8% | +10.9 / +30.3% |
| Coauthor-Physics | +1.6 / +1.7% | +1.5 / +1.5% | +22.4 / +37.7% | +32.9 / +52.6% | +33.2 / +84.2% | −21.9 / −26.7% | +4.8 / +8.6% | +7.5 / +16.9% |

Table 15: Improvement of GEBM in o.o.d. detection AUC-ROC over a vanilla logit-based EBM (GCN-EBM) in an inductive setting.

| | Model | LoC *(last)* AUC-ROC↑ (Alea. / Epi.) | LoC *(last)* Acc.↑ | LoC *(hetero.)* AUC-ROC↑ (Alea. / Epi.) | LoC *(hetero.)* Acc.↑ | Ber($\hat{p}$) *(near)* AUC-ROC↑ (Alea. / Epi.) | Ber($\hat{p}$) *(near)* Acc.↑ | Ber(0.5) AUC-ROC↑ (Alea. / Epi.) | Ber(0.5) Acc.↑ | $\mathcal{N}(0,1)$ *(far)* AUC-ROC↑ (Alea. / Epi.) | $\mathcal{N}(0,1)$ *(far)* Acc.↑ | Page Rank AUC-ROC↑ (Alea. / Epi.) | Page Rank Acc.↑ | Homophily AUC-ROC↑ (Alea. / Epi.) | Homophily Acc.↑ |
|---|---|---|---|---|---|---|---|---|---|---|---|---|---|---|---|
| CoraML | GCN-MCD | $91.6_{\pm0.6}/50.0_{\pm2.3}$ | $90.4_{\pm2.0}$ | $89.2_{\pm1.3}/49.9_{\pm2.6}$ | $87.4_{\pm1.3}$ | $77.2_{\pm2.9}/50.4_{\pm2.7}$ | $75.6_{\pm2.3}$ | $75.6_{\pm2.4}/49.4_{\pm3.4}$ | $75.6_{\pm2.3}$ | $45.3_{\pm2.2}/50.3_{\pm2.7}$ | $75.6_{\pm2.3}$ | $80.4_{\pm1.7}/50.0_{\pm3.4}$ | $82.3_{\pm1.5}$ | $85.6_{\pm1.5}/50.7_{\pm3.7}$ | $91.4_{\pm1.8}$ |
| | GCN-DE | $88.4_{\pm1.2}/76.6_{\pm1.9}$ | $87.2_{\pm1.7}$ | $85.3_{\pm1.1}/72.3_{\pm1.9}$ | $82.0_{\pm2.3}$ | $67.8_{\pm2.5}/67.8_{\pm3.1}$ | $71.9_{\pm2.7}$ | $64.7_{\pm3.4}/70.8_{\pm3.9}$ | $71.9_{\pm2.7}$ | $40.5_{\pm3.1}/58.4_{\pm3.2}$ | $71.9_{\pm2.7}$ | $77.7_{\pm2.8}/71.6_{\pm2.4}$ | $79.3_{\pm2.1}$ | $80.7_{\pm1.7}/68.6_{\pm2.0}$ | $85.6_{\pm1.9}$ |
| | GCN-MCD+DE | $88.5_{\pm1.1}/76.4_{\pm1.8}$ | $87.3_{\pm1.6}$ | $85.4_{\pm1.4}/72.5_{\pm2.2}$ | $83.0_{\pm2.2}$ | $67.7_{\pm2.8}/68.2_{\pm2.3}$ | $72.1_{\pm3.1}$ | $63.9_{\pm3.0}/70.6_{\pm3.9}$ | $72.1_{\pm3.1}$ | $41.3_{\pm3.4}/59.1_{\pm3.3}$ | $72.1_{\pm3.1}$ | $77.8_{\pm2.7}/70.9_{\pm2.8}$ | $78.9_{\pm2.2}$ | $80.8_{\pm2.0}/68.7_{\pm2.9}$ | $85.9_{\pm2.1}$ |
| | GCN-BNN | $90.1_{\pm1.1}/89.1_{\pm1.4}$ | $89.0_{\pm1.4}$ | $87.3_{\pm2.1}/86.5_{\pm1.9}$ | $84.9_{\pm3.0}$ | $77.0_{\pm3.9}/71.7_{\pm7.6}$ | $73.1_{\pm2.3}$ | $75.7_{\pm3.7}/74.8_{\pm5.1}$ | $73.1_{\pm2.3}$ | $48.7_{\pm2.6}/55.0_{\pm3.0}$ | $73.1_{\pm2.3}$ | $81.3_{\pm2.2}/80.7_{\pm2.8}$ | $78.9_{\pm1.8}$ | $87.0_{\pm1.1}/85.6_{\pm1.7}$ | $88.6_{\pm1.9}$ |
| | GCN-Ens | $91.6_{\pm0.5}/75.9_{\pm4.9}$ | $90.4_{\pm2.0}$ | $89.1_{\pm1.4}/66.5_{\pm5.3}$ | $87.3_{\pm1.0}$ | $74.9_{\pm2.7}/48.1_{\pm4.4}$ | $75.9_{\pm2.3}$ | $76.1_{\pm2.3}/56.7_{\pm2.1}$ | $75.9_{\pm2.3}$ | $45.2_{\pm2.7}/54.9_{\pm2.4}$ | $75.9_{\pm2.3}$ | $80.2_{\pm1.8}/66.4_{\pm3.9}$ | $82.7_{\pm1.3}$ | $86.2_{\pm1.6}/60.9_{\pm4.6}$ | $91.5_{\pm1.8}$ |
| | GPN | $89.6_{\pm1.3}/89.6_{\pm1.4}$ | $88.5_{\pm1.4}$ | $87.1_{\pm2.9}/82.4_{\pm3.0}$ | $86.4_{\pm2.7}$ | $81.8_{\pm1.7}/67.7_{\pm4.2}$ | $72.8_{\pm3.0}$ | $80.5_{\pm3.2}/68.5_{\pm5.1}$ | $72.8_{\pm3.0}$ | $82.7_{\pm1.9}/67.5_{\pm3.8}$ | $72.8_{\pm3.0}$ | $81.9_{\pm2.5}/67.5_{\pm6.4}$ | $78.4_{\pm3.2}$ | $79.6_{\pm5.2}/67.6_{\pm6.7}$ | $87.6_{\pm2.0}$ |
| | SGCN | $91.2_{\pm0.9}/89.3_{\pm1.6}$ | $90.2_{\pm1.9}$ | $90.9_{\pm1.5}/88.8_{\pm3.3}$ | $89.9_{\pm1.4}$ | $69.2_{\pm8.4}/51.2_{\pm10.9}$ | $36.2_{\pm24.1}$ | $65.4_{\pm9.5}/54.4_{\pm10.0}$ | $36.2_{\pm24.1}$ | $49.2_{\pm4.3}/49.6_{\pm3.9}$ | $36.2_{\pm24.1}$ | $79.4_{\pm4.6}/69.4_{\pm14.0}$ | $74.8_{\pm22.3}$ | $87.3_{\pm1.6}/80.4_{\pm2.6}$ | $90.0_{\pm1.4}$ |
| | GCN-EBM | $91.6_{\pm0.6}/91.4_{\pm0.7}$ | $90.3_{\pm1.9}$ | $89.1_{\pm1.2}/89.0_{\pm1.4}$ | $87.4_{\pm1.3}$ | $75.2_{\pm3.5}/74.4_{\pm4.1}$ | $75.8_{\pm2.2}$ | $76.4_{\pm2.3}/75.5_{\pm2.9}$ | $75.8_{\pm2.2}$ | $46.8_{\pm4.6}/44.6_{\pm4.0}$ | $75.8_{\pm2.2}$ | $80.4_{\pm1.8}/78.3_{\pm1.7}$ | $82.6_{\pm1.5}$ | $86.0_{\pm1.6}/84.8_{\pm1.9}$ | $91.2_{\pm1.6}$ |
| | GCN-HEAT | $91.6_{\pm0.6}/88.9_{\pm1.9}$ | $90.3_{\pm1.9}$ | $89.1_{\pm1.2}/86.8_{\pm2.9}$ | $87.4_{\pm1.3}$ | $75.2_{\pm3.5}/68.6_{\pm5.2}$ | $75.8_{\pm2.2}$ | $76.4_{\pm2.3}/71.4_{\pm4.7}$ | $75.8_{\pm2.2}$ | $46.8_{\pm4.6}/58.7_{\pm3.5}$ | $75.8_{\pm2.2}$ | $80.4_{\pm1.8}/75.7_{\pm2.6}$ | $82.6_{\pm1.5}$ | $86.0_{\pm1.6}/81.7_{\pm2.6}$ | $91.2_{\pm1.6}$ |
| | GCNSafe | $91.6_{\pm0.6}/93.1_{\pm0.7}$ | $90.3_{\pm1.9}$ | $89.1_{\pm1.2}/90.4_{\pm1.1}$ | $87.4_{\pm1.3}$ | $75.2_{\pm3.5}/76.6_{\pm4.7}$ | $75.8_{\pm2.2}$ | $76.4_{\pm2.3}/74.5_{\pm2.8}$ | $75.8_{\pm2.2}$ | $46.8_{\pm4.6}/46.1_{\pm3.7}$ | $75.8_{\pm2.2}$ | $80.4_{\pm1.8}/78.7_{\pm1.8}$ | $82.6_{\pm1.5}$ | $86.0_{\pm1.6}/84.5_{\pm1.7}$ | $91.2_{\pm1.6}$ |
| | **GCN-GEBM** | $91.6_{\pm0.6}/92.6_{\pm0.8}$ | $90.3_{\pm1.9}$ | $89.1_{\pm1.2}/89.6_{\pm1.6}$ | $87.4_{\pm1.3}$ | $75.2_{\pm3.5}/73.0_{\pm4.6}$ | $75.8_{\pm2.2}$ | $76.4_{\pm2.3}/70.6_{\pm3.1}$ | $75.8_{\pm2.2}$ | $46.8_{\pm4.6}/59.0_{\pm3.5}$ | $75.8_{\pm2.2}$ | $80.4_{\pm1.8}/77.2_{\pm2.4}$ | $82.6_{\pm1.5}$ | $86.0_{\pm1.6}/83.1_{\pm1.9}$ | $91.2_{\pm1.6}$ |
| CoraML-LLM | GCN-MCD | $90.0_{\pm1.1}/50.0_{\pm2.5}$ | $90.1_{\pm1.0}$ | $89.2_{\pm2.9}/49.0_{\pm2.2}$ | $89.5_{\pm2.3}$ | n.a./n.a. | n.a. | $77.5_{\pm2.0}/51.0_{\pm3.2}$ | $76.9_{\pm1.4}$ | $51.7_{\pm1.6}/50.1_{\pm2.1}$ | $76.9_{\pm1.4}$ | $85.2_{\pm1.6}/50.6_{\pm3.3}$ | $84.0_{\pm1.5}$ | $87.2_{\pm1.1}/50.7_{\pm2.6}$ | $91.0_{\pm0.7}$ |
| | GCN-DE | $88.4_{\pm1.1}/76.3_{\pm3.0}$ | $87.8_{\pm1.2}$ | $87.4_{\pm1.9}/74.8_{\pm3.0}$ | $87.9_{\pm2.2}$ | n.a./n.a. | n.a. | $76.5_{\pm1.6}/54.4_{\pm3.4}$ | $75.6_{\pm1.5}$ | $48.7_{\pm2.3}/64.7_{\pm2.4}$ | $75.6_{\pm1.5}$ | $83.8_{\pm1.4}/76.4_{\pm2.2}$ | $81.8_{\pm2.2}$ | $84.2_{\pm1.9}/75.2_{\pm2.4}$ | $88.5_{\pm1.1}$ |
| | GCN-MCD+DE | $88.0_{\pm1.6}/76.8_{\pm2.7}$ | $88.1_{\pm1.4}$ | $87.1_{\pm2.7}/75.1_{\pm3.1}$ | $87.9_{\pm1.9}$ | n.a./n.a. | n.a. | $75.7_{\pm1.6}/55.8_{\pm3.7}$ | $75.2_{\pm2.1}$ | $50.1_{\pm2.4}/67.3_{\pm2.7}$ | $75.2_{\pm2.1}$ | $83.0_{\pm1.6}/75.6_{\pm2.1}$ | $81.3_{\pm1.8}$ | $84.2_{\pm1.5}/76.2_{\pm2.0}$ | $88.7_{\pm1.0}$ |
| | GCN-BNN | $89.8_{\pm1.8}/85.7_{\pm2.5}$ | $90.0_{\pm1.2}$ | $88.8_{\pm2.0}/86.2_{\pm1.8}$ | $89.3_{\pm2.2}$ | n.a./n.a. | n.a. | $77.9_{\pm1.1}/57.1_{\pm6.2}$ | $76.8_{\pm1.5}$ | $54.0_{\pm2.7}/67.6_{\pm2.3}$ | $76.8_{\pm1.5}$ | $85.7_{\pm2.1}/83.9_{\pm2.1}$ | $83.7_{\pm1.6}$ | $88.5_{\pm1.4}/86.8_{\pm1.5}$ | $90.8_{\pm0.9}$ |
| | GCN-Ens | $90.0_{\pm1.0}/81.7_{\pm4.0}$ | $90.1_{\pm1.0}$ | $89.2_{\pm2.7}/79.5_{\pm5.0}$ | $89.5_{\pm2.4}$ | n.a./n.a. | n.a. | $77.7_{\pm1.7}/44.2_{\pm5.3}$ | $77.3_{\pm1.3}$ | $52.2_{\pm2.0}/67.0_{\pm1.4}$ | $77.3_{\pm1.3}$ | $85.2_{\pm1.7}/73.4_{\pm3.0}$ | $84.2_{\pm1.2}$ | $87.1_{\pm1.0}/79.2_{\pm2.5}$ | $91.3_{\pm0.6}$ |
| | GPN | $88.3_{\pm1.8}/90.1_{\pm1.3}$ | $89.4_{\pm0.9}$ | $88.5_{\pm2.3}/85.1_{\pm2.8}$ | $88.1_{\pm2.4}$ | n.a./n.a. | n.a. | $82.5_{\pm2.6}/71.5_{\pm3.5}$ | $75.8_{\pm2.3}$ | $83.0_{\pm2.5}/72.3_{\pm3.7}$ | $75.8_{\pm2.3}$ | $84.6_{\pm2.0}/73.9_{\pm2.4}$ | $81.3_{\pm1.9}$ | $82.5_{\pm2.6}/72.4_{\pm4.3}$ | $90.8_{\pm1.2}$ |
| | SGCN | $90.5_{\pm1.0}/87.6_{\pm1.5}$ | $88.9_{\pm0.8}$ | $88.2_{\pm1.9}/85.4_{\pm3.8}$ | $89.8_{\pm1.9}$ | n.a./n.a. | n.a. | $78.5_{\pm6.6}/61.7_{\pm11.5}$ | $68.1_{\pm18.7}$ | $51.5_{\pm3.2}/48.8_{\pm3.1}$ | $68.1_{\pm18.7}$ | $84.2_{\pm1.8}/72.0_{\pm11.5}$ | $78.9_{\pm14.6}$ | $82.7_{\pm7.1}/73.7_{\pm14.0}$ | $80.2_{\pm20.4}$ |
| | GCN-EBM | $89.9_{\pm1.0}/89.2_{\pm1.2}$ | $90.2_{\pm0.9}$ | $89.1_{\pm2.8}/88.5_{\pm2.8}$ | $89.4_{\pm2.3}$ | n.a./n.a. | n.a. | $78.1_{\pm1.4}/76.0_{\pm1.9}$ | $77.1_{\pm1.3}$ | $51.3_{\pm3.3}/48.3_{\pm3.3}$ | $77.1_{\pm1.3}$ | $85.2_{\pm1.7}/80.9_{\pm1.5}$ | $83.7_{\pm1.5}$ | $87.1_{\pm1.0}/84.9_{\pm0.8}$ | $91.1_{\pm0.7}$ |
| | GCN-HEAT | $89.9_{\pm1.0}/87.9_{\pm2.3}$ | $90.2_{\pm0.9}$ | $89.1_{\pm2.8}/85.6_{\pm4.1}$ | $89.4_{\pm2.3}$ | n.a./n.a. | n.a. | $78.1_{\pm1.4}/71.5_{\pm3.7}$ | $77.1_{\pm1.3}$ | $51.3_{\pm3.3}/62.5_{\pm3.7}$ | $77.1_{\pm1.3}$ | $85.2_{\pm1.7}/73.9_{\pm3.7}$ | $83.7_{\pm1.5}$ | $87.1_{\pm1.0}/80.3_{\pm2.5}$ | $91.1_{\pm0.7}$ |
| | GCNSafe | $89.9_{\pm1.0}/91.8_{\pm1.1}$ | $90.2_{\pm0.9}$ | $89.1_{\pm2.8}/90.9_{\pm1.0}$ | $89.4_{\pm2.3}$ | n.a./n.a. | n.a. | $78.1_{\pm1.4}/80.5_{\pm0.8}$ | $77.1_{\pm1.3}$ | $51.3_{\pm3.3}/49.2_{\pm3.0}$ | $77.1_{\pm1.3}$ | $85.2_{\pm1.7}/82.9_{\pm1.1}$ | $83.7_{\pm1.5}$ | $87.1_{\pm1.0}/85.5_{\pm1.4}$ | $91.1_{\pm0.7}$ |
| | **GCN-GEBM** | $89.9_{\pm1.0}/90.4_{\pm1.2}$ | $90.2_{\pm0.9}$ | $89.1_{\pm2.8}/89.8_{\pm2.4}$ | $89.4_{\pm2.3}$ | n.a./n.a. | n.a. | $78.1_{\pm1.4}/71.7_{\pm2.8}$ | $77.1_{\pm1.3}$ | $51.3_{\pm3.3}/62.6_{\pm4.1}$ | $77.1_{\pm1.3}$ | $85.2_{\pm1.7}/81.3_{\pm2.1}$ | $83.7_{\pm1.5}$ | $87.1_{\pm1.0}/82.6_{\pm1.9}$ | $91.1_{\pm0.7}$ |
| Citeseer | GCN-MCD | $89.5_{\pm1.3}/50.2_{\pm2.3}$ | $84.8_{\pm1.2}$ | $90.5_{\pm1.2}/50.2_{\pm2.2}$ | $90.9_{\pm0.6}$ | $73.7_{\pm1.3}/50.4_{\pm1.7}$ | $76.4_{\pm1.5}$ | $72.6_{\pm3.2}/49.8_{\pm2.1}$ | $76.4_{\pm1.5}$ | $43.1_{\pm1.6}/50.5_{\pm2.5}$ | $76.4_{\pm1.5}$ | $83.4_{\pm0.7}/50.1_{\pm2.5}$ | $78.4_{\pm1.1}$ | $82.6_{\pm1.4}/50.4_{\pm3.0}$ | $77.8_{\pm1.9}$ |
| | GCN-DE | $84.8_{\pm2.1}/76.8_{\pm2.3}$ | $81.0_{\pm1.2}$ | $87.1_{\pm1.4}/74.9_{\pm2.9}$ | $86.4_{\pm1.0}$ | $68.2_{\pm2.2}/70.7_{\pm2.6}$ | $72.9_{\pm1.6}$ | $62.7_{\pm2.6}/78.3_{\pm1.6}$ | $72.9_{\pm1.6}$ | $39.3_{\pm2.2}/61.9_{\pm2.2}$ | $72.9_{\pm1.6}$ | $80.0_{\pm1.6}/76.3_{\pm1.6}$ | $75.6_{\pm1.4}$ | $80.8_{\pm1.6}/75.5_{\pm2.5}$ | $74.3_{\pm1.8}$ |
| | GCN-MCD+DE | $85.3_{\pm1.9}/77.6_{\pm2.7}$ | $81.3_{\pm1.2}$ | $87.5_{\pm1.2}/75.2_{\pm2.0}$ | $87.1_{\pm1.1}$ | $66.8_{\pm2.1}/70.4_{\pm1.8}$ | $73.0_{\pm1.5}$ | $63.4_{\pm2.8}/77.7_{\pm1.8}$ | $73.0_{\pm1.5}$ | $39.6_{\pm2.7}/63.0_{\pm2.2}$ | $73.0_{\pm1.5}$ | $80.1_{\pm1.5}/75.4_{\pm2.9}$ | $75.0_{\pm1.8}$ | $81.0_{\pm1.3}/75.9_{\pm1.7}$ | $74.6_{\pm1.6}$ |
| | GCN-BNN | $89.2_{\pm1.4}/90.1_{\pm1.1}$ | $84.5_{\pm1.3}$ | $90.1_{\pm1.4}/89.9_{\pm1.3}$ | $90.3_{\pm1.0}$ | $75.8_{\pm2.7}/72.1_{\pm7.0}$ | $75.8_{\pm1.5}$ | $68.6_{\pm6.7}/78.6_{\pm3.8}$ | $75.8_{\pm1.5}$ | $43.5_{\pm2.1}/59.7_{\pm2.7}$ | $75.8_{\pm1.5}$ | $84.2_{\pm0.9}/86.0_{\pm0.9}$ | $77.8_{\pm0.6}$ | $84.4_{\pm1.3}/85.8_{\pm1.1}$ | $77.1_{\pm2.0}$ |
| | GCN-Ens | $89.6_{\pm1.0}/81.0_{\pm4.0}$ | $84.8_{\pm1.0}$ | $90.4_{\pm1.4}/81.3_{\pm3.0}$ | $91.1_{\pm0.5}$ | $74.2_{\pm1.7}/58.6_{\pm3.8}$ | $76.9_{\pm1.6}$ | $73.2_{\pm2.0}/74.9_{\pm3.0}$ | $76.9_{\pm1.6}$ | $44.4_{\pm2.0}/58.9_{\pm2.0}$ | $76.9_{\pm1.6}$ | $83.2_{\pm0.8}/79.3_{\pm1.5}$ | $78.6_{\pm1.0}$ | $82.6_{\pm1.4}/76.2_{\pm2.8}$ | $77.7_{\pm1.8}$ |
| | GPN | $90.9_{\pm1.7}/89.6_{\pm1.7}$ | $85.6_{\pm1.6}$ | $92.6_{\pm1.0}/87.3_{\pm2.7}$ | $91.6_{\pm1.2}$ | $81.8_{\pm2.6}/77.4_{\pm2.9}$ | $76.7_{\pm1.9}$ | $82.7_{\pm6.5}/82.9_{\pm5.1}$ | $76.7_{\pm1.9}$ | $86.8_{\pm2.6}/80.4_{\pm2.5}$ | $76.7_{\pm1.9}$ | $86.8_{\pm1.6}/80.1_{\pm2.6}$ | $78.3_{\pm2.3}$ | $85.9_{\pm3.3}/79.5_{\pm4.2}$ | $77.4_{\pm2.2}$ |
| | SGCN | $89.4_{\pm1.4}/88.1_{\pm2.0}$ | $85.0_{\pm1.1}$ | $90.4_{\pm1.3}/88.2_{\pm1.6}$ | $90.4_{\pm0.7}$ | $75.9_{\pm1.8}/73.5_{\pm2.0}$ | $75.8_{\pm1.4}$ | $79.6_{\pm2.1}/77.4_{\pm2.3}$ | $75.8_{\pm1.4}$ | $43.9_{\pm2.0}/41.7_{\pm2.5}$ | $75.8_{\pm1.4}$ | $83.6_{\pm0.9}/82.1_{\pm1.3}$ | $79.3_{\pm1.1}$ | $83.6_{\pm1.3}/82.4_{\pm1.5}$ | $76.6_{\pm2.0}$ |
| | GCN-EBM | $89.5_{\pm1.1}/89.5_{\pm1.3}$ | $84.8_{\pm1.0}$ | $90.5_{\pm1.2}/90.1_{\pm1.5}$ | $90.9_{\pm0.7}$ | $75.2_{\pm1.5}/74.6_{\pm2.3}$ | $76.6_{\pm1.5}$ | $73.9_{\pm3.2}/65.6_{\pm4.8}$ | $76.6_{\pm1.5}$ | $45.3_{\pm2.1}/40.6_{\pm2.3}$ | $76.6_{\pm1.5}$ | $83.2_{\pm1.0}/81.8_{\pm1.3}$ | $78.7_{\pm1.1}$ | $82.8_{\pm1.2}/82.1_{\pm1.6}$ | $77.5_{\pm1.9}$ |
| | GCN-HEAT | $89.5_{\pm1.1}/87.9_{\pm1.8}$ | $84.8_{\pm1.0}$ | $90.5_{\pm1.2}/88.4_{\pm2.3}$ | $90.9_{\pm0.7}$ | $75.2_{\pm1.5}/69.5_{\pm2.1}$ | $76.6_{\pm1.5}$ | $73.9_{\pm3.2}/75.7_{\pm1.9}$ | $76.6_{\pm1.5}$ | $45.3_{\pm2.1}/66.5_{\pm2.1}$ | $76.6_{\pm1.5}$ | $83.2_{\pm1.0}/81.0_{\pm1.2}$ | $78.7_{\pm1.1}$ | $82.8_{\pm1.2}/81.1_{\pm1.8}$ | $77.5_{\pm1.9}$ |
| | GCNSafe | $89.5_{\pm1.1}/91.8_{\pm0.9}$ | $84.8_{\pm1.0}$ | $90.5_{\pm1.2}/91.9_{\pm1.2}$ | $90.9_{\pm0.7}$ | $75.2_{\pm1.5}/80.5_{\pm1.9}$ | $76.6_{\pm1.5}$ | $73.9_{\pm3.2}/75.0_{\pm3.8}$ | $76.6_{\pm1.5}$ | $45.3_{\pm2.1}/41.8_{\pm2.9}$ | $76.6_{\pm1.5}$ | $83.2_{\pm1.0}/85.7_{\pm1.1}$ | $78.7_{\pm1.1}$ | $82.8_{\pm1.2}/86.1_{\pm1.3}$ | $77.5_{\pm1.9}$ |
| | **GCN-GEBM** | $89.5_{\pm1.1}/91.0_{\pm0.9}$ | $84.8_{\pm1.0}$ | $90.5_{\pm1.2}/90.7_{\pm1.0}$ | $90.9_{\pm0.7}$ | $75.2_{\pm1.5}/73.7_{\pm1.9}$ | $76.6_{\pm1.5}$ | $73.9_{\pm3.2}/74.9_{\pm2.1}$ | $76.6_{\pm1.5}$ | $45.3_{\pm2.1}/59.0_{\pm2.8}$ | $76.6_{\pm1.5}$ | $83.2_{\pm1.0}/82.9_{\pm1.1}$ | $78.7_{\pm1.1}$ | $82.8_{\pm1.2}/83.5_{\pm1.3}$ | $77.5_{\pm1.9}$ |
| PubMed | GCN-MCD | $71.1_{\pm1.6}/50.1_{\pm0.7}$ | $92.3_{\pm1.8}$ | $73.9_{\pm1.5}/49.9_{\pm1.0}$ | $82.2_{\pm0.8}$ | n.a./n.a. | n.a. | $69.3_{\pm1.3}/50.4_{\pm0.9}$ | $76.9_{\pm1.5}$ | $49.7_{\pm0.8}/50.1_{\pm0.7}$ | $76.9_{\pm1.5}$ | $73.7_{\pm1.4}/49.9_{\pm0.9}$ | $78.7_{\pm0.9}$ | $74.3_{\pm1.1}/50.2_{\pm0.9}$ | $83.3_{\pm1.9}$ |
| | GCN-DE | $69.7_{\pm2.2}/62.3_{\pm1.7}$ | $91.8_{\pm1.9}$ | $71.0_{\pm2.0}/63.8_{\pm2.9}$ | $79.2_{\pm2.0}$ | n.a./n.a. | n.a. | $63.1_{\pm1.5}/66.1_{\pm2.3}$ | $74.9_{\pm1.7}$ | $46.6_{\pm0.9}/52.9_{\pm1.1}$ | $74.9_{\pm1.7}$ | $71.6_{\pm1.3}/65.1_{\pm1.8}$ | $76.8_{\pm1.6}$ | $72.1_{\pm1.2}/65.6_{\pm1.9}$ | $80.9_{\pm2.3}$ |
| | GCN-MCD+DE | $69.5_{\pm2.1}/62.3_{\pm2.0}$ | $91.7_{\pm1.9}$ | $71.1_{\pm2.4}/64.2_{\pm3.1}$ | $79.3_{\pm1.6}$ | n.a./n.a. | n.a. | $62.7_{\pm1.8}/65.3_{\pm3.4}$ | $74.9_{\pm1.6}$ | $46.6_{\pm1.1}/53.3_{\pm1.1}$ | $74.9_{\pm1.6}$ | $71.3_{\pm1.4}/64.8_{\pm1.6}$ | $77.1_{\pm1.3}$ | $72.2_{\pm1.3}/66.0_{\pm1.9}$ | $80.7_{\pm2.4}$ |
| | GCN-BNN | $70.0_{\pm3.0}/69.5_{\pm3.3}$ | $92.2_{\pm1.7}$ | $73.5_{\pm2.1}/72.5_{\pm3.0}$ | $81.7_{\pm3.1}$ | n.a./n.a. | n.a. | $69.9_{\pm2.4}/65.6_{\pm4.3}$ | $76.4_{\pm1.1}$ | $49.7_{\pm1.1}/55.4_{\pm1.1}$ | $76.4_{\pm1.1}$ | $73.6_{\pm2.7}/73.7_{\pm3.3}$ | $77.1_{\pm3.3}$ | $75.7_{\pm1.2}/75.9_{\pm1.3}$ | $82.8_{\pm2.2}$ |
| | GCN-Ens | $71.2_{\pm1.7}/53.0_{\pm6.5}$ | $92.3_{\pm1.8}$ | $74.0_{\pm1.5}/47.8_{\pm6.6}$ | $82.2_{\pm0.8}$ | n.a./n.a. | n.a. | $69.4_{\pm1.6}/61.6_{\pm5.1}$ | $76.9_{\pm1.5}$ | $49.6_{\pm0.7}/55.4_{\pm1.0}$ | $76.9_{\pm1.5}$ | $73.9_{\pm1.3}/55.9_{\pm5.6}$ | $78.7_{\pm0.8}$ | $74.1_{\pm1.2}/55.0_{\pm6.0}$ | $83.3_{\pm1.8}$ |
| | GPN | $68.1_{\pm3.7}/65.7_{\pm2.6}$ | $92.9_{\pm0.8}$ | $74.4_{\pm2.8}/70.5_{\pm3.0}$ | $85.0_{\pm1.3}$ | n.a./n.a. | n.a. | $73.8_{\pm1.5}/62.4_{\pm2.3}$ | $76.9_{\pm1.1}$ | $74.2_{\pm1.3}/62.4_{\pm2.4}$ | $76.9_{\pm1.1}$ | $77.0_{\pm1.4}/66.5_{\pm2.0}$ | $80.2_{\pm1.1}$ | $72.7_{\pm2.9}/62.7_{\pm3.9}$ | $86.2_{\pm1.5}$ |
| | SGCN | $69.8_{\pm1.7}/69.6_{\pm1.8}$ | $92.7_{\pm1.1}$ | $73.4_{\pm2.0}/70.3_{\pm2.3}$ | $83.8_{\pm1.2}$ | n.a./n.a. | n.a. | $71.5_{\pm2.7}/64.5_{\pm2.7}$ | $77.2_{\pm1.4}$ | $49.4_{\pm1.5}/48.3_{\pm1.3}$ | $77.2_{\pm1.4}$ | $75.4_{\pm1.5}/68.1_{\pm2.3}$ | $80.8_{\pm1.0}$ | $75.4_{\pm1.1}/70.3_{\pm2.7}$ | $83.7_{\pm1.9}$ |
| | GCN-EBM | $71.0_{\pm1.6}/70.8_{\pm1.7}$ | $92.3_{\pm1.8}$ | $74.0_{\pm1.5}/73.7_{\pm1.5}$ | $82.2_{\pm0.8}$ | n.a./n.a. | n.a. | $69.0_{\pm1.4}/63.3_{\pm2.2}$ | $76.9_{\pm1.4}$ | $49.1_{\pm0.7}/48.7_{\pm0.8}$ | $76.9_{\pm1.4}$ | $73.8_{\pm1.3}/69.5_{\pm1.8}$ | $78.6_{\pm0.9}$ | $74.1_{\pm1.3}/70.8_{\pm1.5}$ | $83.3_{\pm2.0}$ |
| | GCN-HEAT | $71.0_{\pm1.6}/69.2_{\pm2.0}$ | $92.3_{\pm1.8}$ | $74.0_{\pm1.5}/68.9_{\pm3.4}$ | $82.2_{\pm0.8}$ | n.a./n.a. | n.a. | $69.0_{\pm1.4}/62.4_{\pm2.6}$ | $76.9_{\pm1.4}$ | $49.1_{\pm0.7}/56.9_{\pm3.0}$ | $76.9_{\pm1.4}$ | $73.8_{\pm1.3}/67.3_{\pm2.7}$ | $78.6_{\pm0.9}$ | $74.1_{\pm1.3}/69.8_{\pm2.0}$ | $83.3_{\pm2.0}$ |
| | GCNSafe | $71.0_{\pm1.6}/71.0_{\pm2.5}$ | $92.3_{\pm1.8}$ | $74.0_{\pm1.5}/75.8_{\pm1.6}$ | $82.2_{\pm0.8}$ | n.a./n.a. | n.a. | $69.0_{\pm1.4}/67.0_{\pm2.5}$ | $76.9_{\pm1.4}$ | $49.1_{\pm0.7}/49.3_{\pm0.9}$ | $76.9_{\pm1.4}$ | $73.8_{\pm1.3}/70.6_{\pm1.2}$ | $78.6_{\pm0.9}$ | $74.1_{\pm1.3}/71.3_{\pm1.3}$ | $83.3_{\pm2.0}$ |
| | **GCN-GEBM** | $71.0_{\pm1.6}/73.9_{\pm1.4}$ | $92.3_{\pm1.8}$ | $74.0_{\pm1.5}/71.7_{\pm2.8}$ | $82.2_{\pm0.8}$ | n.a./n.a. | n.a. | $69.0_{\pm1.4}/63.5_{\pm2.3}$ | $76.9_{\pm1.4}$ | $49.1_{\pm0.7}/53.2_{\pm1.9}$ | $76.9_{\pm1.4}$ | $73.8_{\pm1.3}/66.3_{\pm3.6}$ | $78.6_{\pm0.9}$ | $74.1_{\pm1.3}/71.7_{\pm1.9}$ | $83.3_{\pm2.0}$ |
| Amazon Computers | GCN-MCD | $77.1_{\pm5.1}/49.8_{\pm0.9}$ | $88.8_{\pm1.8}$ | $87.8_{\pm3.8}/49.7_{\pm1.0}$ | $94.2_{\pm1.1}$ | $66.7_{\pm1.9}/50.3_{\pm1.3}$ | $80.2_{\pm2.5}$ | $65.1_{\pm3.4}/49.7_{\pm1.2}$ | $80.2_{\pm2.5}$ | $49.1_{\pm3.7}/49.7_{\pm1.1}$ | $80.2_{\pm2.5}$ | $74.7_{\pm2.9}/50.2_{\pm1.4}$ | $81.0_{\pm1.5}$ | $75.3_{\pm0.5}/49.9_{\pm1.2}$ | $91.2_{\pm1.3}$ |
| | GCN-DE | $78.1_{\pm4.3}/78.4_{\pm2.3}$ | $87.9_{\pm1.9}$ | $87.7_{\pm2.9}/82.5_{\pm2.3}$ | $93.4_{\pm1.3}$ | $63.3_{\pm2.1}/68.9_{\pm1.5}$ | $78.8_{\pm2.5}$ | $62.2_{\pm3.8}/62.4_{\pm10.7}$ | $78.8_{\pm2.5}$ | $45.6_{\pm2.5}/65.5_{\pm4.5}$ | $78.8_{\pm2.5}$ | $73.5_{\pm2.7}/73.7_{\pm1.9}$ | $80.4_{\pm1.3}$ | $74.0_{\pm0.9}/70.9_{\pm0.9}$ | $89.5_{\pm1.4}$ |
| | GCN-MCD+DE | $77.8_{\pm3.5}/78.2_{\pm1.8}$ | $88.2_{\pm1.7}$ | $87.4_{\pm3.2}/82.3_{\pm2.2}$ | $93.4_{\pm1.4}$ | $63.7_{\pm1.9}/68.7_{\pm2.0}$ | $78.5_{\pm2.8}$ | $62.1_{\pm4.3}/62.7_{\pm10.0}$ | $78.5_{\pm2.8}$ | $46.2_{\pm2.4}/65.0_{\pm5.1}$ | $78.5_{\pm2.8}$ | $73.6_{\pm2.9}/73.8_{\pm2.1}$ | $80.5_{\pm1.2}$ | $74.1_{\pm1.4}/70.7_{\pm1.4}$ | $89.8_{\pm1.4}$ |
| | GCN-BNN | $78.7_{\pm7.0}/82.1_{\pm7.3}$ | $84.3_{\pm9.1}$ | $87.3_{\pm3.6}/89.0_{\pm4.0}$ | $93.4_{\pm1.9}$ | $69.8_{\pm3.8}/68.1_{\pm4.7}$ | $78.3_{\pm2.5}$ | $68.0_{\pm5.5}/66.2_{\pm7.6}$ | $78.3_{\pm2.5}$ | $46.7_{\pm2.1}/62.6_{\pm4.3}$ | $78.3_{\pm2.5}$ | $75.2_{\pm3.3}/78.3_{\pm3.7}$ | $77.7_{\pm3.2}$ | $77.3_{\pm1.4}/80.0_{\pm1.4}$ | $90.8_{\pm1.4}$ |
| | GCN-Ens | $77.2_{\pm5.1}/79.9_{\pm4.7}$ | $88.9_{\pm1.9}$ | $87.7_{\pm3.8}/84.1_{\pm3.9}$ | $94.2_{\pm1.1}$ | $67.1_{\pm1.3}/53.7_{\pm3.6}$ | $80.4_{\pm2.5}$ | $65.1_{\pm3.6}/51.0_{\pm6.3}$ | $80.4_{\pm2.5}$ | $48.3_{\pm2.2}/65.4_{\pm4.1}$ | $80.4_{\pm2.5}$ | $74.6_{\pm2.9}/69.3_{\pm3.9}$ | $81.2_{\pm1.3}$ | $75.3_{\pm0.6}/74.5_{\pm1.5}$ | $91.3_{\pm1.2}$ |
| | GPN | $83.9_{\pm3.5}/76.8_{\pm3.8}$ | $86.4_{\pm1.9}$ | $85.1_{\pm4.5}/83.7_{\pm2.6}$ | $94.3_{\pm1.1}$ | $75.7_{\pm5.0}/62.2_{\pm6.1}$ | $74.4_{\pm14.7}$ | $75.2_{\pm5.8}/62.2_{\pm6.0}$ | $74.4_{\pm14.7}$ | $76.3_{\pm6.3}/62.9_{\pm5.8}$ | $74.4_{\pm14.7}$ | $77.6_{\pm2.5}/67.2_{\pm2.4}$ | $78.5_{\pm2.2}$ | $79.9_{\pm3.6}/67.0_{\pm5.2}$ | $92.4_{\pm1.2}$ |
| | SGCN | $50.3_{\pm15.0}/60.0_{\pm14.6}$ | $39.5_{\pm27.2}$ | $56.2_{\pm19.6}/68.6_{\pm13.6}$ | $59.1_{\pm26.4}$ | $52.9_{\pm8.9}/47.6_{\pm8.7}$ | $11.7_{\pm12.2}$ | $52.8_{\pm8.8}/47.7_{\pm8.6}$ | $11.7_{\pm12.2}$ | $50.8_{\pm9.2}/51.1_{\pm6.9}$ | $11.7_{\pm12.2}$ | $43.1_{\pm10.5}/57.1_{\pm11.2}$ | $20.0_{\pm14.3}$ | $59.4_{\pm9.9}/43.8_{\pm9.3}$ | $10.2_{\pm9.5}$ |
| | GCN-EBM | $77.5_{\pm5.0}/76.8_{\pm4.5}$ | $88.8_{\pm1.9}$ | $87.5_{\pm3.9}/85.7_{\pm3.4}$ | $94.1_{\pm1.2}$ | $66.8_{\pm1.5}/64.7_{\pm1.5}$ | $80.0_{\pm2.6}$ | $65.4_{\pm4.1}/63.7_{\pm3.0}$ | $80.0_{\pm2.6}$ | $48.4_{\pm2.6}/48.4_{\pm2.3}$ | $80.0_{\pm2.6}$ | $74.9_{\pm2.8}/68.2_{\pm2.6}$ | $81.0_{\pm1.7}$ | $75.2_{\pm0.8}/71.1_{\pm1.1}$ | $91.1_{\pm1.2}$ |
| | GCN-HEAT | $77.5_{\pm5.0}/76.8_{\pm5.7}$ | $88.8_{\pm1.9}$ | $87.5_{\pm3.9}/85.2_{\pm4.5}$ | $94.1_{\pm1.2}$ | $66.8_{\pm1.5}/64.2_{\pm3.8}$ | $80.0_{\pm2.6}$ | $65.4_{\pm4.1}/64.4_{\pm5.4}$ | $80.0_{\pm2.6}$ | $48.4_{\pm2.6}/63.5_{\pm3.0}$ | $80.0_{\pm2.6}$ | $74.9_{\pm2.8}/67.4_{\pm3.2}$ | $81.0_{\pm1.7}$ | $75.2_{\pm0.8}/72.2_{\pm2.3}$ | $91.1_{\pm1.2}$ |
| | GCNSafe | $77.5_{\pm5.0}/87.1_{\pm2.4}$ | $88.8_{\pm1.9}$ | $87.5_{\pm3.9}/87.0_{\pm3.1}$ | $94.1_{\pm1.2}$ | $66.8_{\pm1.5}/67.7_{\pm2.6}$ | $80.0_{\pm2.6}$ | $65.4_{\pm4.1}/69.1_{\pm3.6}$ | $80.0_{\pm2.6}$ | $48.4_{\pm2.6}/50.5_{\pm2.7}$ | $80.0_{\pm2.6}$ | $74.9_{\pm2.8}/74.7_{\pm1.5}$ | $81.0_{\pm1.7}$ | $75.2_{\pm0.8}/72.5_{\pm1.9}$ | $91.1_{\pm1.2}$ |
| | **GCN-GEBM** | $77.5_{\pm5.0}/82.5_{\pm3.0}$ | $88.8_{\pm1.9}$ | $87.5_{\pm3.9}/87.1_{\pm4.4}$ | $94.1_{\pm1.2}$ | $66.8_{\pm1.5}/60.3_{\pm1.9}$ | $80.0_{\pm2.6}$ | $65.4_{\pm4.1}/58.6_{\pm3.8}$ | $80.0_{\pm2.6}$ | $48.4_{\pm2.6}/54.9_{\pm3.1}$ | $80.0_{\pm2.6}$ | $74.9_{\pm2.8}/68.1_{\pm4.1}$ | $81.0_{\pm1.7}$ | $75.2_{\pm0.8}/68.5_{\pm3.1}$ | $91.1_{\pm1.2}$ |
| Amazon Photo | GCN-MCD | $79.0_{\pm3.3}/50.1_{\pm1.1}$ | $92.0_{\pm1.1}$ | $83.4_{\pm2.8}/50.2_{\pm2.0}$ | $95.3_{\pm2.0}$ | $75.8_{\pm2.2}/49.9_{\pm2.4}$ | $89.8_{\pm1.1}$ | $68.0_{\pm3.8}/50.4_{\pm1.6}$ | $89.8_{\pm1.1}$ | $50.9_{\pm3.2}/49.3_{\pm1.6}$ | $89.8_{\pm1.1}$ | $83.6_{\pm2.7}/49.9_{\pm3.0}$ | $91.9_{\pm0.5}$ | $82.3_{\pm0.9}/50.5_{\pm2.8}$ | $97.4_{\pm0.2}$ |
| | GCN-DE | $80.4_{\pm2.4}/66.6_{\pm1.9}$ | $91.7_{\pm1.1}$ | $84.1_{\pm2.6}/76.5_{\pm2.2}$ | $94.6_{\pm2.0}$ | $72.4_{\pm2.2}/71.6_{\pm5.2}$ | $88.5_{\pm1.2}$ | $69.5_{\pm3.6}/55.8_{\pm3.7}$ | $88.5_{\pm1.2}$ | $49.1_{\pm3.0}/65.3_{\pm3.6}$ | $88.5_{\pm1.2}$ | $83.0_{\pm2.3}/79.1_{\pm2.0}$ | $91.3_{\pm0.7}$ | $80.6_{\pm1.2}/75.6_{\pm1.9}$ | $96.6_{\pm0.4}$ |
| | GCN-MCD+DE | $80.6_{\pm2.5}/66.8_{\pm2.2}$ | $91.4_{\pm1.1}$ | $83.6_{\pm2.9}/76.0_{\pm2.0}$ | $94.8_{\pm1.9}$ | $72.7_{\pm2.0}/69.5_{\pm4.5}$ | $88.3_{\pm1.8}$ | $68.5_{\pm3.9}/55.1_{\pm5.1}$ | $88.3_{\pm1.8}$ | $50.4_{\pm2.5}/62.4_{\pm5.9}$ | $88.3_{\pm1.8}$ | $82.8_{\pm2.3}/78.9_{\pm2.3}$ | $91.6_{\pm0.7}$ | $80.3_{\pm1.3}/75.5_{\pm2.5}$ | $96.6_{\pm0.4}$ |
| | GCN-BNN | $77.0_{\pm4.6}/79.4_{\pm5.2}$ | $91.6_{\pm2.0}$ | $87.5_{\pm3.6}/90.1_{\pm3.8}$ | $95.7_{\pm1.9}$ | $78.6_{\pm2.4}/72.4_{\pm4.9}$ | $89.3_{\pm2.1}$ | $72.2_{\pm4.9}/71.5_{\pm5.1}$ | $89.3_{\pm2.1}$ | $53.5_{\pm2.0}/65.6_{\pm3.5}$ | $89.3_{\pm2.1}$ | $84.6_{\pm2.0}/82.7_{\pm2.8}$ | $90.4_{\pm2.7}$ | $85.2_{\pm1.9}/87.0_{\pm1.6}$ | $97.6_{\pm0.4}$ |
| | GCN-Ens | $79.1_{\pm3.3}/82.4_{\pm6.0}$ | $91.9_{\pm1.0}$ | $83.4_{\pm2.7}/84.5_{\pm6.1}$ | $95.3_{\pm1.9}$ | $75.1_{\pm2.4}/48.3_{\pm5.9}$ | $89.9_{\pm1.0}$ | $68.4_{\pm4.0}/57.8_{\pm5.1}$ | $89.9_{\pm1.0}$ | $50.2_{\pm4.3}/63.4_{\pm2.6}$ | $89.9_{\pm1.0}$ | $83.7_{\pm2.8}/72.8_{\pm4.8}$ | $92.1_{\pm0.5}$ | $82.2_{\pm0.9}/79.4_{\pm2.0}$ | $97.5_{\pm0.2}$ |
| | GPN | $80.8_{\pm1.9}/87.1_{\pm3.0}$ | $88.8_{\pm1.3}$ | $91.6_{\pm2.2}/78.2_{\pm3.1}$ | $93.9_{\pm2.2}$ | $79.1_{\pm3.7}/67.7_{\pm4.0}$ | $86.5_{\pm1.8}$ | $78.9_{\pm3.8}/67.6_{\pm4.3}$ | $86.5_{\pm1.8}$ | $80.0_{\pm3.5}/67.4_{\pm4.0}$ | $86.5_{\pm1.8}$ | $82.5_{\pm3.4}/66.2_{\pm4.9}$ | $88.6_{\pm2.0}$ | $84.1_{\pm3.5}/66.4_{\pm5.0}$ | $97.1_{\pm0.5}$ |
| | SGCN | $80.8_{\pm2.3}/80.7_{\pm3.0}$ | $92.1_{\pm0.9}$ | $81.0_{\pm3.0}/80.4_{\pm4.5}$ | $94.3_{\pm2.2}$ | $50.6_{\pm7.0}/51.7_{\pm6.6}$ | $15.4_{\pm10.0}$ | $50.8_{\pm6.2}/51.1_{\pm5.8}$ | $15.4_{\pm10.0}$ | $48.0_{\pm7.9}/53.2_{\pm8.3}$ | $15.4_{\pm10.0}$ | $53.2_{\pm8.9}/52.6_{\pm9.0}$ | $12.2_{\pm8.6}$ | $55.5_{\pm10.0}/51.0_{\pm8.4}$ | $15.0_{\pm5.9}$ |
| | GCN-EBM | $79.1_{\pm3.3}/78.6_{\pm3.9}$ | $91.8_{\pm1.0}$ | $83.4_{\pm2.8}/79.6_{\pm3.2}$ | $95.3_{\pm2.0}$ | $75.7_{\pm2.5}/70.6_{\pm3.3}$ | $89.8_{\pm1.0}$ | $68.0_{\pm3.9}/65.4_{\pm4.1}$ | $89.8_{\pm1.0}$ | $55.2_{\pm2.5}/51.1_{\pm3.1}$ | $89.8_{\pm1.0}$ | $83.7_{\pm2.7}/76.9_{\pm3.7}$ | $91.9_{\pm0.6}$ | $82.0_{\pm1.0}/77.9_{\pm1.0}$ | $97.4_{\pm0.3}$ |
| | GCN-HEAT | $79.1_{\pm3.3}/79.8_{\pm3.8}$ | $91.8_{\pm1.0}$ | $83.4_{\pm2.8}/76.8_{\pm4.5}$ | $95.3_{\pm2.0}$ | $75.7_{\pm2.5}/69.0_{\pm5.3}$ | $89.8_{\pm1.0}$ | $68.0_{\pm3.9}/66.5_{\pm4.6}$ | $89.8_{\pm1.0}$ | $55.2_{\pm2.5}/62.0_{\pm2.5}$ | $89.8_{\pm1.0}$ | $83.7_{\pm2.7}/73.6_{\pm5.0}$ | $91.9_{\pm0.6}$ | $82.0_{\pm1.0}/77.7_{\pm1.9}$ | $97.4_{\pm0.3}$ |
| | GCNSafe | $79.1_{\pm3.3}/78.6_{\pm5.7}$ | $91.8_{\pm1.0}$ | $83.4_{\pm2.8}/86.0_{\pm4.0}$ | $95.3_{\pm2.0}$ | $75.7_{\pm2.5}/66.5_{\pm3.5}$ | $89.8_{\pm1.0}$ | $68.0_{\pm3.9}/67.2_{\pm5.9}$ | $89.8_{\pm1.0}$ | $55.2_{\pm2.5}/57.9_{\pm4.6}$ | $89.8_{\pm1.0}$ | $83.7_{\pm2.7}/74.0_{\pm3.5}$ | $91.9_{\pm0.6}$ | $82.0_{\pm1.0}/79.8_{\pm0.9}$ | $97.4_{\pm0.3}$ |
| | **GCN-GEBM** | $79.1_{\pm3.3}/84.9_{\pm4.5}$ | $91.8_{\pm1.0}$ | $83.4_{\pm2.8}/88.5_{\pm1.7}$ | $95.3_{\pm2.0}$ | $75.7_{\pm2.5}/55.9_{\pm4.2}$ | $89.8_{\pm1.0}$ | $68.0_{\pm3.9}/62.5_{\pm4.8}$ | $89.8_{\pm1.0}$ | $55.2_{\pm2.5}/56.3_{\pm1.8}$ | $89.8_{\pm1.0}$ | $83.7_{\pm2.7}/62.7_{\pm2.6}$ | $91.9_{\pm0.6}$ | $82.0_{\pm1.0}/77.3_{\pm1.9}$ | $97.4_{\pm0.3}$ |
| Coauthor-CS | GCN-MCD | $90.8_{\pm1.2}/50.1_{\pm1.0}$ | $92.3_{\pm0.8}$ | $93.2_{\pm0.8}/49.9_{\pm1.2}$ | $95.1_{\pm0.5}$ | $76.2_{\pm2.4}/50.7_{\pm1.7}$ | $89.9_{\pm1.0}$ | $79.1_{\pm1.3}/49.7_{\pm1.1}$ | $89.9_{\pm1.0}$ | $47.8_{\pm1.4}/50.0_{\pm1.0}$ | $89.9_{\pm1.0}$ | $83.6_{\pm0.9}/50.0_{\pm1.9}$ | $90.8_{\pm0.4}$ | $84.6_{\pm0.6}/49.6_{\pm1.7}$ | $97.9_{\pm0.4}$ |
| | GCN-DE | $90.2_{\pm1.0}/76.7_{\pm1.4}$ | $91.3_{\pm0.7}$ | $91.7_{\pm0.9}/84.1_{\pm0.8}$ | $94.4_{\pm0.7}$ | $73.0_{\pm1.5}/73.1_{\pm1.5}$ | $88.8_{\pm1.1}$ | $70.5_{\pm1.8}/75.6_{\pm1.6}$ | $88.8_{\pm1.1}$ | $42.5_{\pm1.5}/68.5_{\pm1.4}$ | $88.8_{\pm1.1}$ | $80.8_{\pm1.5}/77.6_{\pm1.6}$ | $89.6_{\pm0.6}$ | $80.8_{\pm1.0}/76.4_{\pm0.8}$ | $97.1_{\pm0.5}$ |
| | GCN-MCD+DE | $90.0_{\pm1.3}/76.7_{\pm1.4}$ | $91.4_{\pm0.5}$ | $91.3_{\pm0.8}/83.9_{\pm0.8}$ | $94.3_{\pm0.6}$ | $72.4_{\pm1.8}/72.7_{\pm1.4}$ | $88.5_{\pm1.0}$ | $69.5_{\pm1.9}/75.3_{\pm1.2}$ | $88.5_{\pm1.0}$ | $42.8_{\pm1.7}/67.1_{\pm1.2}$ | $88.5_{\pm1.0}$ | $80.8_{\pm1.3}/77.9_{\pm1.4}$ | $89.6_{\pm0.6}$ | $81.1_{\pm0.9}/76.8_{\pm1.1}$ | $97.1_{\pm0.4}$ |
| | GCN-BNN | $88.4_{\pm1.8}/88.0_{\pm1.8}$ | $90.8_{\pm0.8}$ | $92.4_{\pm0.9}/91.7_{\pm1.4}$ | $94.7_{\pm0.5}$ | $80.6_{\pm1.5}/79.9_{\pm1.5}$ | $89.0_{\pm0.8}$ | $72.6_{\pm4.4}/82.1_{\pm2.3}$ | $89.0_{\pm0.8}$ | $43.0_{\pm2.4}/59.4_{\pm1.4}$ | $89.0_{\pm0.8}$ | $87.0_{\pm0.8}/87.0_{\pm0.9}$ | $89.8_{\pm0.6}$ | $89.2_{\pm0.8}/89.4_{\pm0.8}$ | $97.9_{\pm0.4}$ |
| | GCN-Ens | $90.9_{\pm1.2}/88.6_{\pm1.9}$ | $92.4_{\pm0.8}$ | $93.2_{\pm0.8}/91.2_{\pm1.3}$ | $95.2_{\pm0.5}$ | $76.1_{\pm2.5}/55.8_{\pm2.0}$ | $90.0_{\pm0.9}$ | $79.1_{\pm1.0}/58.7_{\pm1.8}$ | $90.0_{\pm0.9}$ | $46.1_{\pm0.9}/62.4_{\pm1.3}$ | $90.0_{\pm0.9}$ | $83.7_{\pm0.9}/79.2_{\pm1.7}$ | $90.8_{\pm0.4}$ | $84.4_{\pm0.7}/83.5_{\pm0.9}$ | $97.9_{\pm0.4}$ |
| | GPN | $80.1_{\pm2.6}/86.9_{\pm1.5}$ | $86.6_{\pm1.4}$ | $84.9_{\pm2.1}/74.9_{\pm3.6}$ | $91.6_{\pm1.2}$ | $81.9_{\pm1.8}/62.4_{\pm3.9}$ | $81.7_{\pm2.3}$ | $82.5_{\pm1.9}/62.7_{\pm4.1}$ | $81.7_{\pm2.3}$ | $82.5_{\pm1.9}/62.7_{\pm4.1}$ | $81.7_{\pm2.3}$ | $83.5_{\pm1.7}/63.9_{\pm3.0}$ | $83.4_{\pm1.6}$ | $81.5_{\pm3.0}/70.0_{\pm6.2}$ | $97.9_{\pm0.4}$ |
| | SGCN | $74.3_{\pm7.3}/46.3_{\pm13.5}$ | $30.6_{\pm14.7}$ | $69.8_{\pm12.4}/48.0_{\pm12.7}$ | $31.2_{\pm16.2}$ | $55.6_{\pm7.9}/53.8_{\pm6.7}$ | $11.0_{\pm7.0}$ | $53.0_{\pm4.6}/51.6_{\pm3.2}$ | $11.0_{\pm7.0}$ | $50.0_{\pm3.5}/51.6_{\pm6.1}$ | $11.0_{\pm7.0}$ | $56.1_{\pm11.9}/55.9_{\pm13.2}$ | $12.1_{\pm9.1}$ | $54.6_{\pm12.4}/56.8_{\pm12.4}$ | $14.1_{\pm6.5}$ |
| | GCN-EBM | $90.9_{\pm1.2}/90.2_{\pm1.3}$ | $92.3_{\pm0.9}$ | $93.2_{\pm0.8}/91.1_{\pm1.3}$ | $95.1_{\pm0.5}$ | $76.4_{\pm2.6}/74.3_{\pm2.8}$ | $89.8_{\pm0.9}$ | $79.5_{\pm1.3}/78.7_{\pm1.6}$ | $89.8_{\pm0.9}$ | $47.6_{\pm2.4}/41.3_{\pm2.1}$ | $89.8_{\pm0.9}$ | $83.6_{\pm0.9}/74.6_{\pm1.5}$ | $90.9_{\pm0.3}$ | $84.4_{\pm0.9}/78.6_{\pm1.5}$ | $97.8_{\pm0.4}$ |
| | GCN-HEAT | $90.9_{\pm1.2}/89.4_{\pm1.7}$ | $92.3_{\pm0.9}$ | $93.2_{\pm0.8}/89.8_{\pm2.6}$ | $95.1_{\pm0.5}$ | $76.4_{\pm2.6}/71.9_{\pm2.4}$ | $89.8_{\pm0.9}$ | $79.5_{\pm1.3}/73.6_{\pm2.7}$ | $89.8_{\pm0.9}$ | $47.6_{\pm2.4}/65.4_{\pm1.2}$ | $89.8_{\pm0.9}$ | $83.6_{\pm0.9}/74.8_{\pm3.6}$ | $90.9_{\pm0.3}$ | $84.4_{\pm0.9}/85.5_{\pm1.6}$ | $97.8_{\pm0.4}$ |
| | GCNSafe | $90.9_{\pm1.2}/91.9_{\pm1.5}$ | $92.3_{\pm0.9}$ | $93.2_{\pm0.8}/91.5_{\pm1.2}$ | $95.1_{\pm0.5}$ | $76.4_{\pm2.6}/71.9_{\pm2.5}$ | $89.8_{\pm0.9}$ | $79.5_{\pm1.3}/76.1_{\pm1.7}$ | $89.8_{\pm0.9}$ | $47.6_{\pm2.4}/43.1_{\pm2.4}$ | $89.8_{\pm0.9}$ | $83.6_{\pm0.9}/72.7_{\pm1.9}$ | $90.9_{\pm0.3}$ | $84.4_{\pm0.9}/75.6_{\pm1.4}$ | $97.8_{\pm0.4}$ |
| | **GCN-GEBM** | $90.9_{\pm1.2}/92.4_{\pm1.3}$ | $92.3_{\pm0.9}$ | $93.2_{\pm0.8}/92.1_{\pm1.3}$ | $95.1_{\pm0.5}$ | $76.4_{\pm2.6}/67.5_{\pm2.5}$ | $89.8_{\pm0.9}$ | $79.5_{\pm1.3}/70.1_{\pm3.1}$ | $89.8_{\pm0.9}$ | $47.6_{\pm2.4}/63.2_{\pm1.8}$ | $89.8_{\pm0.9}$ | $83.6_{\pm0.9}/70.3_{\pm2.3}$ | $90.9_{\pm0.3}$ | $84.4_{\pm0.9}/78.2_{\pm1.9}$ | $97.8_{\pm0.4}$ |
| Coauthor-Physics | GCN-MCD | $93.4_{\pm1.5}/49.7_{\pm1.0}$ | $96.7_{\pm0.5}$ | $96.9_{\pm0.4}/50.1_{\pm0.8}$ | $98.3_{\pm0.2}$ | $86.5_{\pm0.8}/49.9_{\pm1.1}$ | $91.9_{\pm1.2}$ | $81.8_{\pm2.0}/49.9_{\pm1.3}$ | $91.9_{\pm1.2}$ | $53.9_{\pm2.1}/49.9_{\pm0.6}$ | $91.9_{\pm1.2}$ | $88.1_{\pm1.0}/49.9_{\pm1.4}$ | $93.4_{\pm1.0}$ | $91.7_{\pm0.3}/50.3_{\pm1.2}$ | $98.0_{\pm0.2}$ |
| | GCN-DE | $93.1_{\pm1.5}/87.2_{\pm1.7}$ | $96.4_{\pm0.7}$ | $96.3_{\pm0.4}/92.4_{\pm0.5}$ | $98.1_{\pm0.2}$ | $83.5_{\pm1.0}/81.9_{\pm1.0}$ | $91.1_{\pm1.2}$ | $78.7_{\pm2.0}/78.6_{\pm1.7}$ | $91.1_{\pm1.2}$ | $51.0_{\pm2.1}/61.1_{\pm2.7}$ | $91.1_{\pm1.2}$ | $87.1_{\pm1.2}/84.9_{\pm1.3}$ | $93.1_{\pm1.0}$ | $90.2_{\pm0.5}/86.3_{\pm0.8}$ | $97.7_{\pm0.2}$ |
| | GCN-MCD+DE | $93.3_{\pm1.3}/87.5_{\pm1.3}$ | $96.5_{\pm0.6}$ | $96.5_{\pm0.5}/92.5_{\pm0.6}$ | $98.2_{\pm0.2}$ | $83.2_{\pm1.0}/82.3_{\pm1.0}$ | $91.3_{\pm1.5}$ | $79.1_{\pm2.5}/79.1_{\pm2.0}$ | $91.3_{\pm1.5}$ | $50.9_{\pm1.2}/61.8_{\pm2.0}$ | $91.3_{\pm1.5}$ | $87.1_{\pm1.2}/84.7_{\pm1.1}$ | $93.1_{\pm0.9}$ | $90.2_{\pm0.5}/86.3_{\pm0.8}$ | $97.7_{\pm0.2}$ |
| | GCN-BNN | $90.9_{\pm3.2}/89.4_{\pm5.2}$ | $96.6_{\pm1.1}$ | $95.3_{\pm1.4}/95.3_{\pm1.5}$ | $98.1_{\pm0.3}$ | $87.3_{\pm1.4}/85.7_{\pm2.2}$ | $91.7_{\pm1.4}$ | $76.9_{\pm6.0}/80.1_{\pm5.4}$ | $91.7_{\pm1.4}$ | $51.3_{\pm6.2}/54.4_{\pm4.3}$ | $91.7_{\pm1.4}$ | $89.2_{\pm1.5}/90.0_{\pm1.2}$ | $93.4_{\pm0.9}$ | $92.9_{\pm0.6}/92.4_{\pm1.1}$ | $98.4_{\pm0.2}$ |
| | GCN-Ens | $93.4_{\pm1.5}/88.6_{\pm3.2}$ | $96.8_{\pm0.5}$ | $96.9_{\pm0.4}/94.3_{\pm1.1}$ | $98.3_{\pm0.2}$ | $86.5_{\pm0.6}/63.2_{\pm4.3}$ | $92.0_{\pm1.2}$ | $82.1_{\pm1.9}/54.7_{\pm4.3}$ | $92.0_{\pm1.2}$ | $54.7_{\pm1.7}/60.5_{\pm2.6}$ | $92.0_{\pm1.2}$ | $88.0_{\pm1.0}/82.6_{\pm5.0}$ | $93.3_{\pm1.0}$ | $91.7_{\pm0.3}/88.9_{\pm0.8}$ | $98.0_{\pm0.2}$ |
| | GPN | $87.0_{\pm2.4}/89.1_{\pm2.8}$ | $96.8_{\pm0.4}$ | $85.9_{\pm2.3}/92.4_{\pm1.4}$ | $97.7_{\pm0.3}$ | $88.4_{\pm1.3}/69.6_{\pm5.0}$ | $90.3_{\pm1.4}$ | $87.8_{\pm1.6}/68.2_{\pm5.6}$ | $90.3_{\pm1.4}$ | $87.7_{\pm1.6}/68.1_{\pm5.6}$ | $90.3_{\pm1.4}$ | $88.1_{\pm1.4}/69.3_{\pm5.3}$ | $92.3_{\pm1.0}$ | $85.1_{\pm7.3}/76.8_{\pm12.0}$ | $98.8_{\pm0.2}$ |
| | SGCN | $93.0_{\pm1.2}/91.2_{\pm2.1}$ | $96.9_{\pm0.4}$ | $96.7_{\pm0.5}/95.3_{\pm1.6}$ | $98.3_{\pm0.2}$ | $87.0_{\pm1.1}/78.0_{\pm4.0}$ | $91.3_{\pm1.4}$ | $82.3_{\pm2.0}/69.8_{\pm5.5}$ | $91.3_{\pm1.4}$ | $51.7_{\pm1.9}/45.9_{\pm2.1}$ | $91.3_{\pm1.4}$ | $85.4_{\pm1.3}/76.7_{\pm3.2}$ | $93.3_{\pm0.7}$ | $90.1_{\pm0.9}/83.7_{\pm3.6}$ | $97.5_{\pm0.6}$ |
| | GCN-EBM | $93.4_{\pm1.5}/93.2_{\pm1.8}$ | $96.8_{\pm0.5}$ | $96.9_{\pm0.3}/96.9_{\pm0.5}$ | $98.3_{\pm0.2}$ | $86.3_{\pm0.9}/84.1_{\pm1.7}$ | $92.0_{\pm1.2}$ | $82.2_{\pm2.0}/79.3_{\pm2.4}$ | $92.0_{\pm1.2}$ | $54.5_{\pm2.1}/49.9_{\pm1.7}$ | $92.0_{\pm1.2}$ | $88.0_{\pm1.1}/84.1_{\pm1.4}$ | $93.4_{\pm1.0}$ | $91.7_{\pm0.4}/89.3_{\pm0.6}$ | $98.0_{\pm0.2}$ |
| | GCN-HEAT | $93.4_{\pm1.5}/91.5_{\pm1.4}$ | $96.8_{\pm0.5}$ | $96.9_{\pm0.3}/95.2_{\pm0.6}$ | $98.3_{\pm0.2}$ | $86.3_{\pm0.9}/81.2_{\pm2.2}$ | $92.0_{\pm1.2}$ | $82.2_{\pm2.0}/79.2_{\pm3.2}$ | $92.0_{\pm1.2}$ | $54.5_{\pm2.1}/59.3_{\pm2.5}$ | $92.0_{\pm1.2}$ | $88.0_{\pm1.1}/84.5_{\pm3.1}$ | $93.4_{\pm1.0}$ | $91.7_{\pm0.4}/89.7_{\pm0.9}$ | $98.0_{\pm0.2}$ |
| | GCNSafe | $93.4_{\pm1.5}/95.0_{\pm1.2}$ | $96.8_{\pm0.5}$ | $96.9_{\pm0.3}/97.8_{\pm0.3}$ | $98.3_{\pm0.2}$ | $86.3_{\pm0.9}/84.2_{\pm2.0}$ | $92.0_{\pm1.2}$ | $82.2_{\pm2.0}/80.4_{\pm2.0}$ | $92.0_{\pm1.2}$ | $54.5_{\pm2.1}/51.8_{\pm2.7}$ | $92.0_{\pm1.2}$ | $88.0_{\pm1.1}/84.5_{\pm0.8}$ | $93.4_{\pm1.0}$ | $91.7_{\pm0.4}/86.8_{\pm0.5}$ | $98.0_{\pm0.2}$ |
| | **GCN-GEBM** | $93.4_{\pm1.5}/95.0_{\pm1.5}$ | $96.8_{\pm0.5}$ | $96.9_{\pm0.3}/98.0_{\pm0.5}$ | $98.3_{\pm0.2}$ | $86.3_{\pm0.9}/80.5_{\pm2.5}$ | $92.0_{\pm1.2}$ | $82.2_{\pm2.0}/74.6_{\pm2.4}$ | $92.0_{\pm1.2}$ | $54.5_{\pm2.1}/53.6_{\pm3.3}$ | $92.0_{\pm1.2}$ | $88.0_{\pm1.1}/86.5_{\pm1.3}$ | $93.4_{\pm1.0}$ | $91.7_{\pm0.4}/89.7_{\pm0.8}$ | $98.0_{\pm0.2}$ |

Table 16: Misclassification detection AUC-ROC (↑) using aleatoric or epistemic uncertainty (best and runner-up) in an inductive setting.

### D.3 Feature Collapse and Spectral Normalization

A recent line of work identifies feature collapse as a limitation in uncertainty estimation for deterministic models on i.i.d. data [46, 69]. Intuitively, feature collapse occurs when out-of-distribution data is mapped to the same regions of the model latent space as in-distribution data. Consequently, density-based uncertainty estimators such as the regularizer of GEBM can not separate both data distributions accurately and falsely attribute high confidence to out-of-distribution examples. To

| Dataset | Model | LoC (last) AUC-ROC↑ (Alea. / Epi.) | LoC (last) Acc.↑ | LoC (hetero.) AUC-ROC↑ (Alea. / Epi.) | LoC (hetero.) Acc.↑ | Ber($\hat{p}$) (near) AUC-ROC↑ (Alea. / Epi.) | Ber($\hat{p}$) (near) Acc.↑ | Ber(0.5) AUC-ROC↑ (Alea. / Epi.) | Ber(0.5) Acc.↑ | $\mathcal{N}(0,1)$ (far) AUC-ROC↑ (Alea. / Epi.) | $\mathcal{N}(0,1)$ (far) Acc.↑ | Page Rank AUC-ROC↑ (Alea. / Epi.) | Page Rank Acc.↑ | Homophily AUC-ROC↑ (Alea. / Epi.) | Homophily Acc.↑ |
|---|---|---|---|---|---|---|---|---|---|---|---|---|---|---|---|
| CoraML | GCN-MCD | 89.6±1.0/50.1±2.8 | 90.4±0.9 | 86.3±2.2/50.1±1.8 | 85.2±3.3 | 79.1±1.4/50.0±3.2 | 82.0±0.8 | 77.6±2.3/50.1±2.7 | 82.0±0.8 | 47.8±2.4/49.9±2.3 | 82.0±0.8 | 79.1±2.6/49.5±4.2 | 82.6±1.5 | 86.0±1.0/49.8±3.7 | 93.9±0.8 |
| | GCN-DE | 87.2±1.5/75.6±2.2 | 87.3±1.2 | 83.8±2.1/71.3±3.4 | 80.9±2.3 | 70.7±2.3/71.5±2.3 | 78.4±1.9 | 66.3±2.8/71.5±2.6 | 78.4±1.9 | 42.1±2.5/56.4±2.6 | 78.4±1.9 | 77.0±2.2/68.9±2.6 | 79.0±2.3 | 82.2±1.7/73.2±2.6 | 90.6±1.8 |
| | GCN-MCD+DE | 87.5±1.5/76.6±1.9 | 87.6±1.2 | 83.3±2.5/71.9±2.9 | 80.1±2.9 | 69.9±1.8/71.4±2.8 | 78.1±2.2 | 66.0±2.4/72.0±1.7 | 78.1±2.2 | 42.5±3.2/58.7±2.9 | 78.1±2.2 | 76.7±2.2/70.6±1.9 | 79.5±1.7 | 82.9±1.6/73.5±2.6 | 90.8±1.7 |
| | GCN-BNN | 88.0±1.8/88.1±2.1 | 89.0±1.2 | 83.3±3.0/82.6±3.1 | 82.1±3.4 | 79.4±2.3/76.6±3.2 | 80.6±2.0 | 76.0±3.7/76.5±3.8 | 80.6±2.0 | 50.2±2.5/55.5±2.8 | 80.6±2.0 | 78.9±2.7/78.1±2.8 | 80.4±2.3 | 85.8±1.7/85.3±1.8 | 93.1±1.5 |
| | GCN-Ens | 89.8±1.0/76.3±3.2 | 90.3±0.9 | 86.5±2.3/62.3±6.1 | 85.2±3.3 | 78.4±1.9/51.9±3.0 | 81.8±0.8 | 77.3±2.6/58.4±3.4 | 81.8±0.8 | 48.3±2.3/56.1±3.3 | 81.8±0.8 | 78.9±2.4/64.0±3.0 | 82.7±1.5 | 86.1±0.8/75.9±2.7 | 94.0±0.5 |
| | GPN | 88.9±1.2/89.3±1.9 | 90.0±1.0 | 85.3±3.2/82.7±2.4 | 85.5±2.2 | 82.2±2.0/67.8±4.8 | 78.8±2.1 | 80.5±2.8/67.6±5.2 | 78.8±2.1 | 82.1±2.1/67.3±4.6 | 78.8±2.1 | 81.6±2.4/68.7±4.3 | 78.4±1.8 | 83.8±2.3/75.3±3.8 | 91.1±1.8 |
| | SGCN | 89.8±1.1/89.9±1.4 | 91.0±1.2 | 88.0±3.1/88.0±3.2 | 88.5±2.0 | 72.8±10.1/60.2±14.2 | 51.7±30.2 | 68.1±11.5/63.1±13.8 | 51.7±30.2 | 49.2±5.1/47.3±5.4 | 51.7±30.2 | 82.0±1.8/76.7±2.8 | 83.3±1.3 | 86.0±1.1/83.2±1.9 | 94.2±1.0 |
| | GCN-EBM | 89.7±1.0/89.2±1.3 | 90.4±0.9 | 86.4±2.4/85.7±2.8 | 85.0±3.2 | 77.4±1.9/75.4±2.2 | 81.9±0.8 | 78.2±2.5/76.3±3.5 | 81.9±0.8 | 45.6±3.7/41.6±3.5 | 81.9±0.8 | 79.2±2.5/77.1±2.2 | 82.5±1.6 | 85.8±1.0/84.2±1.1 | 94.1±0.6 |
| | GCN-HEAT | 89.7±1.0/87.9±1.2 | 90.4±0.9 | 86.4±2.4/83.3±3.1 | 85.0±3.2 | 77.4±1.9/74.9±2.1 | 81.9±0.8 | 78.2±2.5/74.4±2.0 | 81.9±0.8 | 45.6±3.7/62.0±3.1 | 81.9±0.8 | 79.2±2.5/75.8±3.1 | 82.5±1.6 | 85.8±1.0/83.5±2.3 | 94.1±0.6 |
| | GCNSafe | 89.7±1.0/91.7±1.0 | 90.4±0.9 | 86.4±2.4/88.4±1.2 | 85.0±3.2 | 77.4±1.9/77.7±2.0 | 81.9±0.8 | 78.2±2.5/77.1±2.8 | 81.9±0.8 | 45.6±3.7/43.1±4.4 | 81.9±0.8 | 79.2±2.5/77.7±2.5 | 82.5±1.6 | 85.8±1.0/84.0±1.0 | 94.1±0.6 |
| | **GCN-GEBM** | 89.7±1.0/91.1±1.0 | 90.4±0.9 | 86.4±2.4/87.9±2.2 | 85.0±3.2 | 77.4±1.9/73.4±3.3 | 81.9±0.8 | 78.2±2.5/71.5±2.2 | 81.9±0.8 | 45.6±3.7/57.7±2.8 | 81.9±0.8 | 79.2±2.5/77.7±1.9 | 82.5±1.6 | 85.8±1.0/83.7±1.0 | 94.1±0.6 |
| CoraML-LLM | GCN-MCD | 90.9±1.7/50.5±2.2 | 91.2±0.7 | 89.0±2.1/50.1±2.5 | 90.6±2.0 | n.a./n.a. | n.a. | 79.4±1.4/49.4±2.8 | 82.6±1.5 | 52.3±3.2/50.0±2.6 | 82.6±1.5 | 84.9±0.9/49.5±2.1 | 84.9±2.1 | 88.5±1.6/50.1±3.4 | 94.1±1.1 |
| | GCN-DE | 89.6±1.3/80.3±2.1 | 89.7±1.2 | 87.6±2.8/76.9±3.2 | 88.9±1.7 | n.a./n.a. | n.a. | 76.6±1.7/59.2±3.2 | 79.9±1.8 | 53.3±2.3/68.0±2.0 | 79.9±1.8 | 82.7±1.5/74.8±2.1 | 82.2±2.7 | 86.0±1.7/79.1±2.5 | 91.7±1.4 |
| | GCN-MCD+DE | 89.8±1.2/80.4±1.8 | 89.4±0.9 | 87.8±2.5/76.3±4.0 | 88.9±1.7 | n.a./n.a. | n.a. | 77.2±1.7/58.3±3.2 | 80.3±1.8 | 51.2±3.5/65.9±2.9 | 80.3±1.8 | 82.7±1.3/75.7±1.9 | 82.7±2.6 | 86.1±1.4/78.4±2.2 | 91.4±1.5 |
| | GCN-BNN | 91.0±1.2/88.5±1.7 | 91.4±0.8 | 88.5±2.7/86.1±3.3 | 91.0±2.0 | n.a./n.a. | n.a. | 79.1±1.8/63.2±4.6 | 82.8±1.6 | 55.5±2.7/64.0±2.2 | 82.8±1.6 | 85.3±1.6/83.0±2.1 | 84.0±2.2 | 89.2±1.3/87.7±1.6 | 94.0±1.3 |
| | GCN-Ens | 90.8±1.6/86.8±1.2 | 91.4±0.7 | 89.1±2.2/79.6±4.5 | 90.6±2.3 | n.a./n.a. | n.a. | 79.2±1.0/55.2±6.8 | 82.6±1.5 | 53.8±2.6/68.0±1.6 | 82.6±1.5 | 84.7±0.7/73.6±2.0 | 85.0±2.1 | 88.6±1.5/84.0±1.3 | 94.0±1.1 |
| | GPN | 90.0±1.1/91.2±0.7 | 90.5±1.1 | 88.0±3.0/85.9±2.5 | 88.5±2.2 | n.a./n.a. | n.a. | 82.9±2.0/73.6±2.9 | 79.7±1.8 | 82.7±1.9/73.5±2.4 | 79.7±1.8 | 84.5±1.9/74.9±3.0 | 80.9±2.2 | 85.3±1.5/78.9±3.4 | 92.5±1.0 |
| | SGCN | 91.7±0.9/90.7±1.1 | 90.3±0.9 | 89.6±2.4/88.5±2.8 | 90.6±1.6 | n.a./n.a. | n.a. | 78.8±7.2/62.2±12.1 | 68.7±21.0 | 51.4±4.7/48.6±3.5 | 68.7±21.0 | 84.6±1.3/74.9±4.5 | 82.0±2.7 | 87.0±1.2/82.8±1.7 | 93.1±1.1 |
| | GCN-EBM | 90.7±1.5/90.0±1.7 | 91.4±0.7 | 89.1±2.2/87.9±2.5 | 90.7±2.0 | n.a./n.a. | n.a. | 78.8±1.3/76.6±1.3 | 82.6±1.6 | 55.2±2.9/50.3±3.0 | 82.6±1.6 | 84.8±0.9/81.0±1.1 | 85.0±2.2 | 88.6±1.3/85.9±1.8 | 93.8±1.0 |
| | GCN-HEAT | 90.7±1.5/86.3±4.0 | 91.4±0.7 | 89.1±2.2/84.7±5.0 | 90.7±2.0 | n.a./n.a. | n.a. | 78.8±1.3/73.5±3.1 | 82.6±1.6 | 55.2±2.9/61.5±3.3 | 82.6±1.6 | 84.8±0.9/75.8±4.1 | 85.0±2.2 | 88.6±1.3/83.0±3.1 | 93.8±1.0 |
| | GCNSafe | 90.7±1.5/92.6±1.5 | 91.4±0.7 | 89.1±2.2/91.4±1.1 | 90.7±2.0 | n.a./n.a. | n.a. | 78.8±1.3/80.3±1.2 | 82.6±1.6 | 55.2±2.9/52.8±4.5 | 82.6±1.6 | 84.8±0.9/83.1±0.9 | 85.0±2.2 | 88.6±1.3/86.2±1.2 | 93.8±1.0 |
| | **GCN-GEBM** | 90.7±1.5/91.4±1.1 | 91.4±0.7 | 89.1±2.2/89.0±3.3 | 90.7±2.0 | n.a./n.a. | n.a. | 78.8±1.3/72.8±3.3 | 82.6±1.6 | 55.2±2.9/60.0±2.7 | 82.6±1.6 | 84.8±0.9/79.2±2.5 | 85.0±2.2 | 88.6±1.3/86.3±2.2 | 93.8±1.0 |
| Citeseer | GCN-MCD | 88.0±0.7/49.2±2.0 | 83.5±1.2 | 88.6±1.3/49.3±2.0 | 91.1±0.8 | 74.4±1.7/50.1±2.2 | 82.1±1.0 | 77.1±2.1/50.6±2.4 | 82.1±1.0 | 45.4±2.4/49.5±2.1 | 82.1±1.0 | 82.6±2.1/50.3±2.7 | 82.9±1.1 | 84.6±1.9/50.2±2.8 | 86.3±1.5 |
| | GCN-DE | 85.2±1.6/77.6±2.2 | 80.1±1.7 | 85.9±1.6/74.4±2.4 | 86.4±1.0 | 68.4±1.6/73.5±1.5 | 76.8±1.5 | 64.8±2.7/79.0±1.4 | 76.8±1.5 | 41.4±2.2/61.4±1.9 | 76.8±1.5 | 80.6±2.0/76.1±1.5 | 79.0±1.3 | 81.9±1.4/79.7±1.4 | 81.6±1.6 |
| | GCN-MCD+DE | 84.9±1.8/77.2±2.2 | 80.2±1.7 | 85.5±1.4/75.5±2.0 | 86.2±1.1 | 68.4±2.0/73.6±2.1 | 76.8±1.8 | 65.5±2.3/79.4±1.5 | 76.8±1.8 | 40.4±1.8/61.4±1.8 | 76.8±1.8 | 80.0±2.0/76.6±1.9 | 79.2±1.0 | 82.3±1.6/79.9±1.6 | 81.7±1.5 |
| | GCN-BNN | 87.5±1.1/88.2±1.3 | 82.5±1.4 | 87.5±1.5/88.0±1.6 | 90.2±0.7 | 75.7±1.9/79.6±1.7 | 81.2±1.3 | 74.7±3.3/81.6±3.1 | 81.2±1.3 | 47.8±3.1/60.8±2.1 | 81.2±1.3 | 82.7±2.5/83.8±2.3 | 81.3±1.1 | 84.9±1.0/85.6±0.9 | 86.6±1.1 |
| | GCN-Ens | 87.9±0.5/79.8±2.1 | 83.8±1.1 | 88.7±1.1/81.1±3.3 | 91.2±0.8 | 73.7±1.3/70.6±1.7 | 82.0±1.0 | 75.2±0.9/77.2±2.5 | 82.0±1.0 | 46.5±1.8/58.4±1.6 | 82.0±1.0 | 82.7±1.9/75.9±2.1 | 82.6±0.8 | 84.4±2.0/84.9±1.4 | 86.4±1.6 |
| | GPN | 88.7±2.4/87.3±2.6 | 84.9±1.9 | 91.2±1.7/86.4±2.7 | 90.9±0.9 | 83.5±2.1/80.0±2.6 | 83.3±1.9 | 83.1±5.6/85.9±2.3 | 83.3±1.9 | 88.1±1.6/82.9±2.4 | 83.3±1.9 | 86.3±1.9/78.6±3.6 | 82.1±1.5 | 88.0±1.1/81.9±2.1 | 88.0±1.0 |
| | SGCN | 88.6±1.8/89.7±2.2 | 83.5±1.9 | 89.5±1.1/90.8±1.4 | 90.6±0.6 | 76.6±1.1/73.3±1.3 | 81.5±1.8 | 81.1±3.3/79.0±3.2 | 81.5±1.8 | 43.5±2.2/41.4±2.2 | 81.5±1.8 | 82.7±2.1/80.0±2.1 | 82.5±1.0 | 85.0±1.8/83.5±2.0 | 85.9±1.4 |
| | GCN-EBM | 87.9±0.6/87.9±0.8 | 83.6±1.1 | 88.4±1.2/88.0±1.5 | 91.1±0.9 | 74.2±1.4/72.4±1.7 | 82.0±1.0 | 76.4±2.4/70.5±3.9 | 82.0±1.0 | 45.6±1.4/39.4±1.5 | 82.0±1.0 | 82.7±2.0/81.3±2.2 | 82.4±1.0 | 84.4±1.8/82.7±1.5 | 86.4±1.5 |
| | GCN-HEAT | 87.9±0.6/86.0±1.2 | 83.6±1.1 | 88.4±1.2/86.0±1.6 | 91.1±0.9 | 74.2±1.4/70.4±1.5 | 82.0±1.0 | 76.4±2.4/74.3±1.5 | 82.0±1.0 | 45.6±1.4/65.4±2.9 | 82.0±1.0 | 82.7±2.0/80.5±1.9 | 82.4±1.0 | 84.4±1.8/82.4±2.2 | 86.4±1.5 |
| | GCNSafe | 87.9±0.6/90.7±0.8 | 83.6±1.1 | 88.4±1.2/89.9±1.6 | 91.1±0.9 | 74.2±1.4/79.5±2.3 | 82.0±1.0 | 76.4±2.4/79.6±2.8 | 82.0±1.0 | 45.6±1.4/40.2±2.1 | 82.0±1.0 | 82.7±2.0/84.8±1.6 | 82.4±1.0 | 84.4±1.8/85.9±1.1 | 86.4±1.5 |
| | **GCN-GEBM** | 87.9±0.6/89.6±1.3 | 83.6±1.1 | 88.4±1.2/89.2±1.7 | 91.1±0.9 | 74.2±1.4/73.2±2.7 | 82.0±1.0 | 76.4±2.4/73.5±1.9 | 82.0±1.0 | 45.6±1.4/60.9±2.3 | 82.0±1.0 | 82.7±2.0/81.7±2.2 | 82.4±1.0 | 84.4±1.8/84.1±1.8 | 86.4±1.5 |
| PubMed | GCN-MCD | 71.4±2.6/50.1±0.9 | 91.6±1.6 | 71.2±2.2/50.1±0.9 | 83.7±1.5 | n.a./n.a. | n.a. | 70.3±1.4/50.2±1.0 | 78.1±1.0 | 50.4±0.8/49.8±0.8 | 78.1±1.0 | 73.2±1.5/50.0±1.3 | 77.9±2.6 | 75.6±1.4/49.9±0.8 | 88.9±2.8 |
| | GCN-DE | 70.4±2.4/62.3±3.6 | 91.2±1.7 | 70.6±1.8/65.7±2.4 | 80.5±2.1 | n.a./n.a. | n.a. | 64.0±1.3/66.7±2.2 | 74.9±2.1 | 47.4±0.9/52.7±1.1 | 74.9±2.1 | 70.2±2.3/64.2±3.4 | 76.0±2.6 | 73.0±1.5/68.6±2.0 | 84.8±2.3 |
| | GCN-MCD+DE | 71.2±2.6/62.3±4.2 | 90.8±2.1 | 71.0±2.4/66.0±2.0 | 80.7±1.8 | n.a./n.a. | n.a. | 64.0±1.0/67.4±1.9 | 75.4±1.5 | 47.0±1.1/52.5±0.8 | 75.4±1.5 | 70.5±1.5/65.0±2.6 | 76.3±1.9 | 73.1±1.3/68.9±1.6 | 84.9±1.8 |
| | GCN-BNN | 71.0±4.2/70.6±4.3 | 90.7±1.6 | 71.4±3.2/71.5±3.4 | 84.0±1.7 | n.a./n.a. | n.a. | 70.8±1.8/69.2±2.5 | 78.2±1.6 | 51.0±1.5/53.6±1.3 | 78.2±1.6 | 73.9±2.4/73.8±2.5 | 76.1±4.2 | 76.4±1.9/76.5±1.7 | 89.3±1.9 |
| | GCN-Ens | 71.4±2.6/54.2±3.6 | 91.7±1.5 | 71.2±2.4/47.7±6.8 | 83.8±1.4 | n.a./n.a. | n.a. | 70.0±0.8/59.3±4.0 | 78.1±0.9 | 50.1±1.0/54.7±1.0 | 78.1±0.9 | 73.2±1.4/50.3±5.5 | 77.9±2.6 | 75.6±1.4/63.8±4.1 | 88.9±2.8 |
| | GPN | 71.2±2.6/66.5±5.7 | 92.3±0.8 | 71.4±2.9/71.2±3.0 | 85.6±1.2 | n.a./n.a. | n.a. | 74.4±2.3/64.7±2.4 | 79.8±1.1 | 75.4±1.3/63.7±1.7 | 79.8±1.1 | 77.2±1.2/66.9±1.9 | 80.2±1.4 | 76.4±1.1/68.8±1.1 | 90.3±1.7 |
| | SGCN | 71.3±2.1/71.2±2.3 | 92.1±1.6 | 72.6±2.6/72.1±2.5 | 85.1±0.9 | n.a./n.a. | n.a. | 72.5±2.1/64.4±2.4 | 80.1±0.9 | 49.6±0.9/48.2±1.1 | 80.1±0.9 | 75.2±1.5/67.9±1.9 | 79.3±2.8 | 76.2±1.4/71.5±1.7 | 90.7±1.8 |
| | GCN-EBM | 71.5±2.6/71.4±2.6 | 91.7±1.5 | 71.1±2.4/70.9±2.4 | 83.7±1.5 | n.a./n.a. | n.a. | 70.6±1.2/65.9±2.1 | 78.1±1.0 | 50.3±1.2/49.5±1.2 | 78.1±1.0 | 73.2±1.4/69.2±1.5 | 78.0±2.6 | 75.8±1.3/72.6±1.5 | 88.8±2.7 |
| | GCN-HEAT | 71.5±2.6/68.5±3.3 | 91.7±1.5 | 71.1±2.4/67.0±4.2 | 83.7±1.5 | n.a./n.a. | n.a. | 70.6±1.2/61.5±2.8 | 78.1±1.0 | 50.3±1.2/57.2±1.4 | 78.1±1.0 | 73.2±1.4/65.2±3.4 | 78.0±2.6 | 75.8±1.3/70.4±2.4 | 88.8±2.7 |
| | GCNSafe | 71.5±2.6/72.1±2.5 | 91.7±1.5 | 71.1±2.4/73.7±2.2 | 83.7±1.5 | n.a./n.a. | n.a. | 70.6±1.2/68.1±2.1 | 78.1±1.0 | 50.3±1.2/50.4±1.2 | 78.1±1.0 | 73.2±1.4/70.2±0.8 | 78.0±2.6 | 75.8±1.3/72.8±1.4 | 88.8±2.7 |
| | **GCN-GEBM** | 71.5±2.6/71.1±3.3 | 91.7±1.5 | 71.1±2.4/71.2±2.1 | 83.7±1.5 | n.a./n.a. | n.a. | 70.6±1.2/61.5±2.2 | 78.1±1.0 | 50.3±1.2/54.2±1.9 | 78.1±1.0 | 73.2±1.4/63.3±3.6 | 78.0±2.6 | 75.8±1.3/68.5±3.5 | 88.8±2.7 |
| Amazon Computers | GCN-MCD | 79.5±4.4/49.7±1.0 | 88.0±1.5 | 87.6±2.3/50.0±1.2 | 93.5±2.4 | 67.1±1.4/50.7±1.5 | 81.7±1.1 | 58.9±3.8/50.2±1.0 | 81.7±1.1 | 50.4±3.5/50.5±1.4 | 81.7±1.1 | 71.1±1.2/50.3±1.4 | 82.7±1.1 | 75.2±1.1/49.8±1.3 | 92.3±0.7 |
| | GCN-DE | 79.3±4.1/80.7±2.9 | 87.2±1.2 | 87.6±1.9/83.6±1.9 | 93.0±2.7 | 63.7±1.1/69.3±1.8 | 80.7±1.2 | 57.4±5.7/61.4±8.6 | 80.7±1.2 | 44.9±2.7/65.2±4.3 | 80.7±1.2 | 70.3±1.3/72.3±1.9 | 82.0±1.0 | 74.5±1.3/72.7±1.2 | 90.9±0.8 |
| | GCN-MCD+DE | 79.6±3.6/80.4±2.6 | 87.1±1.2 | 88.0±2.2/84.1±1.9 | 92.8±2.7 | 64.1±1.1/69.9±2.3 | 80.7±1.3 | 58.1±4.3/60.4±8.9 | 80.7±1.3 | 47.0±2.9/66.7±4.4 | 80.7±1.3 | 70.4±1.2/72.3±1.6 | 81.7±1.5 | 75.0±1.2/73.1±1.3 | 91.1±0.7 |
| | GCN-BNN | 75.3±7.3/79.0±7.6 | 85.4±3.2 | 86.0±4.9/87.8±4.8 | 92.3±3.4 | 68.4±3.9/66.7±6.1 | 77.6±2.6 | 63.0±8.2/59.7±8.8 | 77.6±2.6 | 49.1±2.1/66.1±3.6 | 77.6±2.6 | 73.5±2.3/78.2±2.8 | 80.7±2.4 | 77.4±1.4/79.5±1.5 | 90.9±1.6 |
| | GCN-Ens | 79.6±4.6/75.7±5.3 | 88.1±1.7 | 87.5±2.5/86.0±4.1 | 93.5±2.6 | 66.5±1.2/57.2±3.3 | 81.7±0.9 | 59.3±3.8/53.3±4.0 | 81.7±0.9 | 48.9±1.6/63.2±4.7 | 81.7±0.9 | 71.1±1.2/66.2±3.2 | 82.6±1.0 | 75.0±0.9/76.4±1.2 | 92.4±0.6 |
| | GPN | 87.8±2.6/80.4±2.6 | 86.0±1.4 | 88.4±1.8/85.0±1.8 | 94.4±1.1 | 77.9±3.3/66.0±2.9 | 79.5±2.4 | 78.0±3.1/66.3±2.8 | 79.5±2.4 | 79.2±2.8/66.6±2.8 | 79.5±2.4 | 78.8±2.0/67.7±1.7 | 79.3±2.0 | 78.6±7.5/68.5±4.9 | 86.1±14.4 |
| | SGCN | 60.3±20.3/70.5±14.1 | 58.0±29.3 | 80.3±8.3/84.0±6.5 | 92.7±6.5 | 50.3±10.4/51.0±9.8 | 16.1±15.1 | 50.4±9.6/50.8±9.0 | 16.1±15.1 | 46.4±9.6/54.0±7.8 | 16.1±15.1 | 44.5±11.2/55.1±12.3 | 14.9±13.4 | 35.9±14.6/65.2±14.1 | 16.7±10.6 |
| | GCN-EBM | 79.5±4.5/76.9±3.4 | 88.0±1.8 | 87.3±2.4/84.8±2.9 | 93.5±2.5 | 66.5±1.6/63.5±2.1 | 81.9±1.0 | 59.9±4.1/58.4±4.4 | 81.9±1.0 | 49.6±2.0/47.3±2.4 | 81.9±1.0 | 71.0±1.4/66.1±1.8 | 82.6±1.2 | 75.0±1.0/68.7±0.9 | 92.4±0.7 |
| | GCN-HEAT | 79.5±4.5/75.8±4.9 | 88.0±1.8 | 87.3±2.4/84.5±2.8 | 93.5±2.5 | 66.5±1.6/64.3±4.7 | 81.9±1.0 | 59.9±4.1/60.2±6.4 | 81.9±1.0 | 49.6±2.0/59.4±2.4 | 81.9±1.0 | 71.0±1.4/64.9±4.1 | 82.6±1.2 | 75.0±1.0/68.7±3.2 | 92.4±0.7 |
| | GCNSafe | 79.5±4.5/88.4±1.6 | 88.0±1.8 | 87.3±2.4/87.1±2.0 | 93.5±2.5 | 66.5±1.6/68.5±2.5 | 81.9±1.0 | 59.9±4.1/62.5±5.5 | 81.9±1.0 | 49.6±2.0/53.3±5.4 | 81.9±1.0 | 71.0±1.4/72.9±1.5 | 82.6±1.2 | 75.0±1.0/73.1±0.9 | 92.4±0.7 |
| | **GCN-GEBM** | 79.5±4.5/84.8±2.5 | 88.0±1.8 | 87.3±2.4/89.4±1.3 | 93.5±2.5 | 66.5±1.6/56.6±3.2 | 81.9±1.0 | 59.9±4.1/56.6±7.2 | 81.9±1.0 | 49.6±2.0/51.5±3.0 | 81.9±1.0 | 71.0±1.4/66.0±3.1 | 82.6±1.2 | 75.0±1.0/64.1±4.1 | 92.4±0.7 |
| Amazon Photo | GCN-MCD | 79.2±4.0/50.0±1.1 | 92.0±0.9 | 84.0±4.4/49.8±1.5 | 96.3±0.2 | 77.7±2.1/49.9±2.0 | 90.2±2.0 | 72.6±4.4/49.5±1.2 | 90.2±2.0 | 55.2±4.0/49.7±1.5 | 90.2±2.0 | 83.2±1.7/49.5±2.4 | 91.8±0.6 | 84.8±1.5/48.7±2.2 | 97.6±0.7 |
| | GCN-DE | 80.3±3.6/66.3±2.4 | 91.3±1.1 | 83.4±4.1/76.8±3.5 | 95.9±0.4 | 75.2±2.2/71.2±5.9 | 89.5±1.9 | 71.3±4.0/56.6±4.7 | 89.5±1.9 | 52.3±2.6/63.2±5.3 | 89.5±1.9 | 82.4±1.5/77.8±2.4 | 91.2±0.7 | 82.6±1.9/79.5±1.1 | 96.9±0.8 |
| | GCN-MCD+DE | 79.6±4.4/66.3±2.9 | 91.5±1.0 | 83.5±4.2/76.8±3.5 | 95.9±0.5 | 75.4±1.9/71.9±4.6 | 89.5±1.7 | 72.4±4.6/57.3±3.9 | 89.5±1.7 | 52.6±3.5/61.8±5.4 | 89.5±1.7 | 82.1±1.3/77.7±2.1 | 91.2±0.7 | 82.8±1.7/79.5±1.1 | 96.9±0.8 |
| | GCN-BNN | 82.2±5.9/83.5±6.9 | 93.2±0.8 | 87.5±4.5/89.7±4.7 | 96.5±0.4 | 81.4±2.5/75.6±4.3 | 90.0±2.6 | 73.9±4.9/72.2±3.9 | 90.0±2.6 | 54.4±1.9/65.9±2.6 | 90.0±2.6 | 84.2±2.2/80.9±2.9 | 90.7±1.4 | 86.1±2.9/85.7±3.0 | 97.9±0.9 |
| | GCN-Ens | 79.6±4.3/77.7±7.4 | 92.0±0.8 | 83.9±4.3/83.7±4.2 | 96.3±0.2 | 78.4±1.3/53.3±6.1 | 90.4±2.0 | 72.7±4.1/61.3±8.6 | 90.4±2.0 | 52.7±1.3/61.3±2.0 | 90.4±2.0 | 83.3±1.8/71.1±4.3 | 91.9±0.5 | 84.9±1.5/80.8±2.4 | 97.7±0.6 |
| | GPN | 83.3±4.9/87.8±2.4 | 89.5±1.2 | 92.7±1.2/79.9±4.2 | 94.3±1.6 | 82.4±2.5/64.6±5.7 | 86.2±2.6 | 82.3±2.7/64.9±5.5 | 86.2±2.6 | 83.6±2.2/64.6±5.8 | 86.2±2.6 | 81.8±2.3/69.2±4.4 | 88.8±1.9 | 82.8±2.3/72.6±2.9 | 96.4±0.9 |
| | SGCN | 80.1±3.7/81.9±4.1 | 91.9±0.9 | 79.7±4.6/79.5±4.7 | 96.0±1.0 | 53.3±6.9/48.9±6.2 | 11.4±7.5 | 53.5±6.8/48.8±6.0 | 11.4±7.5 | 52.1±10.1/50.1±9.2 | 11.4±7.5 | 74.0±12.0/67.2±15.3 | 64.8±36.0 | 51.7±8.4/50.5±10.9 | 12.2±7.0 |
| | GCN-EBM | 79.6±4.3/77.8±4.2 | 92.0±0.8 | 83.8±4.3/77.5±3.5 | 96.3±0.2 | 78.6±1.7/73.4±2.6 | 90.2±1.9 | 72.7±4.5/70.2±5.4 | 90.2±1.9 | 55.6±2.8/52.0±3.2 | 90.2±1.9 | 83.1±1.8/76.5±2.2 | 91.9±0.6 | 84.8±1.5/76.6±2.5 | 97.7±0.6 |
| | GCN-HEAT | 79.6±4.3/76.7±7.2 | 92.0±0.8 | 83.8±4.3/74.2±8.0 | 96.3±0.2 | 78.6±1.7/70.3±3.9 | 90.2±1.9 | 72.7±4.5/69.9±5.4 | 90.2±1.9 | 55.6±2.8/59.8±4.1 | 90.2±1.9 | 83.1±1.8/72.0±3.2 | 91.9±0.6 | 84.8±1.5/74.8±4.1 | 97.7±0.6 |
| | GCNSafe | 79.6±4.3/78.0±6.1 | 92.0±0.8 | 83.8±4.3/84.2±3.9 | 96.3±0.2 | 78.6±1.7/73.1±4.7 | 90.2±1.9 | 72.7±4.5/72.0±6.3 | 90.2±1.9 | 55.6±2.8/52.2±4.1 | 90.2±1.9 | 83.1±1.8/76.0±2.5 | 91.9±0.6 | 84.8±1.5/79.1±1.9 | 97.7±0.6 |
| | **GCN-GEBM** | 79.6±4.3/85.0±4.6 | 92.0±0.8 | 83.8±4.3/87.6±3.8 | 96.3±0.2 | 78.6±1.7/57.1±3.7 | 90.2±1.9 | 72.7±4.5/64.7±5.8 | 90.2±1.9 | 55.6±2.8/55.9±2.7 | 90.2±1.9 | 83.1±1.8/61.6±5.9 | 91.9±0.6 | 84.8±1.5/67.3±2.9 | 97.7±0.6 |
| Coauthor-CS | GCN-MCD | 90.3±0.8/50.2±1.2 | 93.1±0.3 | 91.2±1.6/49.6±0.9 | 94.8±0.4 | 78.6±1.3/50.1±1.6 | 92.1±0.3 | 78.9±1.3/49.9±1.1 | 92.1±0.3 | 49.4±1.8/49.9±1.2 | 92.1±0.3 | 82.8±1.4/49.9±2.2 | 92.6±0.5 | 86.1±0.5/50.2±2.0 | 98.8±0.4 |
| | GCN-DE | 89.2±1.1/78.6±1.3 | 92.2±0.5 | 89.5±1.7/82.7±1.6 | 94.0±0.5 | 74.5±1.1/74.9±1.4 | 90.9±0.6 | 71.7±2.0/76.9±1.2 | 90.9±0.6 | 43.8±1.2/66.8±1.5 | 90.9±0.6 | 80.6±1.4/77.7±1.3 | 90.9±0.6 | 82.8±1.0/80.0±1.1 | 98.2±0.3 |
| | GCN-MCD+DE | 89.3±1.4/78.4±1.7 | 92.2±0.4 | 89.1±1.9/82.4±1.4 | 93.8±0.6 | 74.7±1.1/75.5±0.6 | 90.8±0.6 | 71.9±1.5/76.4±1.8 | 90.8±0.6 | 44.5±1.3/68.0±1.0 | 90.8±0.6 | 80.7±1.5/77.6±1.5 | 90.9±0.7 | 83.1±0.8/80.1±0.7 | 98.2±0.3 |
| | GCN-BNN | 88.0±2.9/87.2±3.2 | 91.9±0.5 | 89.9±2.4/89.1±2.8 | 94.2±0.8 | 81.6±1.0/81.3±1.4 | 91.2±0.5 | 70.6±7.3/81.3±3.9 | 91.2±0.5 | 42.6±1.8/57.9±2.0 | 91.2±0.5 | 86.7±1.2/87.0±1.1 | 91.7±0.8 | 89.6±1.0/89.5±1.0 | 98.8±0.5 |
| | GCN-Ens | 90.4±0.8/88.4±1.7 | 93.1±0.3 | 91.3±1.6/88.9±1.9 | 94.8±0.4 | 78.1±0.8/63.7±2.1 | 92.2±0.2 | 78.4±0.6/63.8±1.7 | 92.2±0.2 | 46.4±2.1/61.6±1.2 | 92.2±0.2 | 82.9±1.3/78.3±2.0 | 92.7±0.5 | 86.1±0.5/83.8±1.4 | 98.9±0.4 |
| | GPN | 79.6±2.9/87.8±1.7 | 86.7±0.9 | 86.0±2.5/78.0±4.4 | 91.7±1.3 | 82.6±1.4/63.9±1.9 | 85.4±1.7 | 82.7±1.4/64.0±2.0 | 85.4±1.7 | 82.7±1.4/64.0±2.0 | 85.4±1.7 | 83.6±1.5/64.8±3.9 | 86.7±1.5 | 84.5±1.6/65.8±3.8 | 94.8±1.2 |
| | SGCN | 61.8±13.5/59.7±13.9 | 29.0±18.1 | 69.3±8.5/46.2±13.0 | 30.3±13.5 | 57.7±7.4/50.8±8.7 | 14.0±8.5 | 51.6±7.3/53.5±6.7 | 14.0±8.5 | 50.2±3.1/50.2±5.5 | 14.0±8.5 | 57.8±10.2/54.9±10.3 | 15.1±9.2 | 55.7±11.0/57.3±11.6 | 13.1±9.3 |
| | GCN-EBM | 90.3±0.8/89.4±0.9 | 93.1±0.4 | 91.2±1.5/89.1±1.6 | 94.8±0.4 | 78.2±1.2/74.6±1.4 | 92.0±0.2 | 79.0±1.4/76.9±1.7 | 92.0±0.2 | 48.3±1.9/40.8±1.4 | 92.0±0.2 | 82.8±1.2/74.9±1.5 | 92.6±0.5 | 86.1±0.5/77.8±1.0 | 98.9±0.4 |
| | GCN-HEAT | 90.3±0.8/89.3±1.9 | 93.1±0.4 | 91.2±1.5/87.2±4.2 | 94.8±0.4 | 78.2±1.2/73.7±2.0 | 92.0±0.2 | 79.0±1.4/72.9±2.8 | 92.0±0.2 | 48.3±1.9/66.0±1.4 | 92.0±0.2 | 82.8±1.2/75.3±3.0 | 92.6±0.5 | 86.1±0.5/83.2±1.9 | 98.9±0.4 |
| | GCNSafe | 90.3±0.8/91.9±1.1 | 93.1±0.4 | 91.2±1.5/90.8±1.2 | 94.8±0.4 | 78.2±1.2/72.2±1.8 | 92.0±0.2 | 79.0±1.4/75.2±2.0 | 92.0±0.2 | 48.3±1.9/42.3±1.4 | 92.0±0.2 | 82.8±1.2/73.1±1.2 | 92.6±0.5 | 86.1±0.5/75.0±1.2 | 98.9±0.4 |
| | **GCN-GEBM** | 90.3±0.8/92.5±1.2 | 93.1±0.4 | 91.2±1.5/89.8±2.5 | 94.8±0.4 | 78.2±1.2/67.3±1.8 | 92.0±0.2 | 79.0±1.4/68.0±2.7 | 92.0±0.2 | 48.3±1.9/64.1±1.6 | 92.0±0.2 | 82.8±1.2/69.5±3.7 | 92.6±0.5 | 86.1±0.5/75.4±2.7 | 98.9±0.4 |
| Coauthor-Physics | GCN-MCD | 92.3±1.2/50.2±0.7 | 96.6±1.2 | 95.2±1.4/49.9±0.8 | 98.1±0.3 | 86.8±0.6/49.9±1.2 | 92.9±0.7 | 82.4±1.6/50.1±1.0 | 92.9±0.7 | 55.1±2.0/49.9±0.8 | 92.9±0.7 | 88.3±0.6/50.0±1.3 | 94.5±0.8 | 91.1±1.0/49.8±1.4 | 98.8±0.2 |
| | GCN-DE | 92.1±1.0/86.9±1.0 | 96.5±1.0 | 94.5±1.3/90.8±1.2 | 97.9±0.4 | 83.8±0.9/83.6±0.9 | 92.3±0.6 | 80.3±1.8/80.7±1.5 | 92.3±0.6 | 51.4±1.3/62.2±1.4 | 92.3±0.6 | 87.3±1.0/85.0±1.3 | 94.1±0.7 | 89.9±1.1/87.5±0.9 | 98.4±0.3 |
| | GCN-MCD+DE | 91.7±0.9/86.4±1.0 | 96.5±1.0 | 94.8±1.3/91.2±1.1 | 97.9±0.3 | 84.0±1.3/83.5±0.9 | 92.3±0.7 | 79.7±1.6/80.2±1.8 | 92.3±0.7 | 51.5±1.2/62.3±1.8 | 92.3±0.7 | 87.5±0.8/85.1±1.1 | 94.2±0.7 | 89.8±1.1/87.3±1.1 | 98.4±0.3 |
| | GCN-BNN | 90.9±2.0/90.3±2.0 | 96.6±1.0 | 92.3±3.1/92.8±3.1 | 97.9±0.3 | 88.4±1.2/87.0±1.4 | 92.4±0.9 | 73.9±6.9/80.0±6.7 | 92.4±0.9 | 49.7±6.4/55.8±4.8 | 92.4±0.9 | 89.4±2.0/89.5±2.9 | 94.2±1.0 | 92.2±1.2/91.5±1.6 | 98.9±0.3 |
| | GCN-Ens | 92.3±1.4/86.6±5.2 | 96.7±1.2 | 95.3±1.4/91.5±2.7 | 98.1±0.3 | 87.2±0.5/68.5±4.0 | 92.9±0.7 | 82.5±1.3/56.5±5.9 | 92.9±0.7 | 53.9±2.1/58.9±2.3 | 92.9±0.7 | 88.5±0.6/81.4±3.1 | 94.5±0.8 | 91.1±1.0/90.3±1.0 | 98.7±0.3 |
| | GPN | 87.6±2.8/88.8±2.9 | 96.8±0.5 | 85.5±3.3/93.8±2.0 | 97.7±0.3 | 89.1±1.0/73.4±3.0 | 92.3±0.7 | 88.6±1.1/72.6±3.0 | 92.3±0.7 | 88.6±1.1/72.6±3.0 | 92.3±0.7 | 88.5±1.6/71.1±4.9 | 93.3±0.8 | 89.5±1.1/74.8±3.5 | 98.1±0.4 |
| | SGCN | 92.0±1.2/90.1±3.0 | 96.5±1.0 | 95.3±1.0/94.4±1.7 | 98.1±0.4 | 87.0±1.8/78.6±4.6 | 92.4±0.9 | 83.4±2.1/72.5±4.7 | 92.4±0.9 | 51.8±1.4/45.6±2.3 | 92.4±0.9 | 85.4±1.5/78.1±3.3 | 94.0±0.7 | 89.2±1.5/84.0±2.8 | 98.6±0.4 |
| | GCN-EBM | 92.3±1.1/91.7±1.8 | 96.7±1.1 | 95.2±1.4/95.3±1.3 | 98.1±0.3 | 87.0±0.6/84.1±1.6 | 92.8±0.7 | 82.6±1.6/79.3±2.6 | 92.8±0.7 | 54.0±1.6/48.4±1.7 | 92.8±0.7 | 88.4±0.7/84.5±1.2 | 94.5±0.7 | 91.1±0.9/88.0±1.3 | 98.7±0.2 |
| | GCN-HEAT | 92.3±1.1/89.2±2.7 | 96.7±1.1 | 95.2±1.4/93.3±1.7 | 98.1±0.3 | 87.0±0.6/81.6±1.8 | 92.8±0.7 | 82.6±1.6/77.4±3.8 | 92.8±0.7 | 54.0±1.6/59.4±1.8 | 92.8±0.7 | 88.4±0.7/85.0±2.0 | 94.5±0.7 | 91.1±0.9/87.7±1.7 | 98.7±0.2 |
| | GCNSafe | 92.3±1.1/94.3±1.3 | 96.7±1.1 | 95.2±1.4/97.1±0.8 | 98.1±0.3 | 87.0±0.6/84.0±1.5 | 92.8±0.7 | 82.6±1.6/81.8±2.1 | 92.8±0.7 | 54.0±1.6/49.9±2.6 | 92.8±0.7 | 88.4±0.7/85.0±0.8 | 94.5±0.7 | 91.1±0.9/86.1±1.1 | 98.7±0.2 |
| | **GCN-GEBM** | 92.3±1.1/94.0±1.6 | 96.7±1.1 | 95.2±1.4/97.3±0.8 | 98.1±0.3 | 87.0±0.6/80.4±2.0 | 92.8±0.7 | 82.6±1.6/76.3±2.8 | 92.8±0.7 | 54.0±1.6/55.2±2.7 | 92.8±0.7 | 88.4±0.7/87.7±1.1 | 94.5±0.7 | 91.1±0.9/89.3±0.9 | 98.7±0.2 |

Table 17: Misclassification detection AUC-ROC (↑) using aleatoric or epistemic uncertainty (best and runner-up) in a transductive setting.

mitigate this issue, spectral normalization has been proposed to enforce Lipschitz smoothness in a classifier [46, 47]. We investigate if this regularization benefits GEBM as well in Appendix D.3.

In contrast to work on i.i.d. data, we do not find spectral normalization consistently leading to improved out-of-distribution detection. We provide two possible explanations: First, feature collapse has not been observed to occur for graph problems. The diffusion process at the backbone of many GNNs may mitigate the issue. Second, spectral normalization enforces Lipschitz smoothness in terms of the $L_2$ norm of the input features of a node and its representation. This notion neglects the structural influence of other nodes and, therefore, may not be suitable to quantify feature collapse in non-i.i.d. domains such as graphs.

| | Model | LoC (last) AUC-PR↑ (Alea. / Epi.) | Acc.↑ | LoC (hetero.) AUC-PR↑ (Alea. / Epi.) | Acc.↑ | Ber($\hat{p}$) (near) AUC-PR↑ (Alea. / Epi.) | Acc.↑ | Ber(0.5) AUC-PR↑ (Alea. / Epi.) | Acc.↑ | $\mathcal{N}(0,1)$ (far) AUC-PR↑ (Alea. / Epi.) | Acc.↑ | Page Rank AUC-PR↑ (Alea. / Epi.) | Acc.↑ | Homophily AUC-PR↑ (Alea. / Epi.) | Acc.↑ |
|---|---|---|---|---|---|---|---|---|---|---|---|---|---|---|---|
| CoraML | GCN-MCD | $90.5_{\pm0.7}/51.5_{\pm2.2}$ | $90.4_{\pm2.0}$ | $84.7_{\pm2.2}/42.1_{\pm2.4}$ | $87.4_{\pm1.3}$ | $51.8_{\pm3.5}/29.3_{\pm2.6}$ | $75.6_{\pm2.3}$ | $54.0_{\pm4.6}/33.8_{\pm4.7}$ | $75.6_{\pm2.3}$ | $74.5_{\pm2.1}/76.7_{\pm1.6}$ | $75.6_{\pm2.3}$ | $43.1_{\pm2.7}/20.7_{\pm3.5}$ | $82.3_{\pm1.5}$ | $52.3_{\pm5.1}/23.7_{\pm3.5}$ | $91.4_{\pm1.8}$ |
| | GCN-DE | $87.3_{\pm1.9}/72.5_{\pm2.5}$ | $87.2_{\pm1.7}$ | $80.6_{\pm2.4}/60.8_{\pm2.9}$ | $82.0_{\pm2.3}$ | $50.6_{\pm4.2}/52.8_{\pm4.8}$ | $71.9_{\pm2.7}$ | $58.4_{\pm7.1}/65.8_{\pm6.6}$ | $71.9_{\pm2.7}$ | $70.9_{\pm2.4}/81.7_{\pm2.3}$ | $71.9_{\pm2.7}$ | $47.1_{\pm5.1}/38.4_{\pm4.1}$ | $79.3_{\pm2.1}$ | $51.6_{\pm3.8}/34.7_{\pm3.6}$ | $85.6_{\pm1.9}$ |
| | GCN-MCD+DE | $87.6_{\pm1.8}/72.6_{\pm2.2}$ | $87.3_{\pm1.6}$ | $81.1_{\pm1.9}/61.0_{\pm3.3}$ | $83.0_{\pm2.2}$ | $50.6_{\pm4.5}/53.1_{\pm3.4}$ | $72.1_{\pm3.1}$ | $57.3_{\pm6.5}/64.6_{\pm7.4}$ | $72.1_{\pm3.1}$ | $72.2_{\pm2.3}/83.1_{\pm2.4}$ | $72.1_{\pm3.1}$ | $47.2_{\pm5.5}/37.9_{\pm4.5}$ | $78.9_{\pm2.2}$ | $51.2_{\pm4.4}/34.6_{\pm5.0}$ | $85.9_{\pm2.1}$ |
| | GCN-BNN | $88.4_{\pm1.8}/85.8_{\pm2.4}$ | $89.0_{\pm1.4}$ | $81.0_{\pm4.0}/78.1_{\pm3.7}$ | $84.9_{\pm3.0}$ | $54.9_{\pm7.0}/46.7_{\pm7.1}$ | $73.1_{\pm2.3}$ | $70.4_{\pm3.1}/69.3_{\pm7.5}$ | $73.1_{\pm2.3}$ | $78.3_{\pm2.3}/81.7_{\pm2.0}$ | $73.1_{\pm2.3}$ | $49.0_{\pm4.0}/47.1_{\pm6.0}$ | $78.9_{\pm1.8}$ | $56.8_{\pm4.6}/48.6_{\pm4.9}$ | $88.6_{\pm1.9}$ |
| | GCN-Ens | $90.4_{\pm0.8}/74.2_{\pm2.5}$ | $90.4_{\pm2.0}$ | $84.5_{\pm2.3}/57.5_{\pm5.3}$ | $87.3_{\pm1.0}$ | $47.2_{\pm6.7}/25.6_{\pm2.3}$ | $75.9_{\pm2.3}$ | $54.8_{\pm4.3}/35.9_{\pm4.1}$ | $75.9_{\pm2.3}$ | $74.4_{\pm2.0}/79.8_{\pm1.7}$ | $75.9_{\pm2.3}$ | $42.8_{\pm2.4}/27.2_{\pm4.2}$ | $82.7_{\pm1.3}$ | $55.2_{\pm5.1}/24.1_{\pm2.3}$ | $91.5_{\pm1.8}$ |
| | GPN | $87.2_{\pm2.5}/88.3_{\pm1.1}$ | $88.5_{\pm1.4}$ | $82.5_{\pm3.8}/74.5_{\pm4.0}$ | $86.4_{\pm2.7}$ | $56.5_{\pm5.4}/39.8_{\pm7.5}$ | $72.8_{\pm3.0}$ | $56.2_{\pm6.8}/41.4_{\pm11.1}$ | $72.8_{\pm3.0}$ | $59.6_{\pm5.1}/40.1_{\pm5.2}$ | $72.8_{\pm3.0}$ | $53.6_{\pm4.6}/37.1_{\pm10.1}$ | $78.4_{\pm3.2}$ | $59.5_{\pm8.2}/42.9_{\pm8.8}$ | $87.6_{\pm2.0}$ |
| | SGCN | $89.5_{\pm1.8}/87.5_{\pm2.2}$ | $90.2_{\pm1.9}$ | $86.5_{\pm2.6}/82.0_{\pm7.0}$ | $89.9_{\pm1.4}$ | $82.3_{\pm12.3}/71.4_{\pm18.3}$ | $36.2_{\pm24.1}$ | $83.0_{\pm11.7}/76.9_{\pm16.6}$ | $36.2_{\pm24.1}$ | $82.6_{\pm5.9}/83.0_{\pm5.7}$ | $36.2_{\pm24.1}$ | $52.1_{\pm16.0}/39.0_{\pm15.4}$ | $74.8_{\pm22.3}$ | $58.4_{\pm5.5}/38.9_{\pm5.6}$ | $90.0_{\pm1.4}$ |
| | GCN-EBM | $90.5_{\pm0.8}/90.2_{\pm1.4}$ | $90.3_{\pm1.9}$ | $84.5_{\pm2.2}/84.2_{\pm2.8}$ | $87.4_{\pm1.3}$ | $48.2_{\pm4.9}/47.1_{\pm6.3}$ | $75.8_{\pm2.2}$ | $54.5_{\pm3.7}/53.7_{\pm3.9}$ | $75.8_{\pm2.2}$ | $75.8_{\pm2.8}/75.0_{\pm2.9}$ | $75.8_{\pm2.2}$ | $43.6_{\pm2.7}/40.9_{\pm4.1}$ | $82.6_{\pm1.5}$ | $53.8_{\pm4.8}/51.8_{\pm5.3}$ | $91.2_{\pm1.6}$ |
| | GCN-HEAT | $90.5_{\pm0.8}/85.5_{\pm4.0}$ | $90.3_{\pm1.9}$ | $84.5_{\pm2.2}/80.1_{\pm5.5}$ | $87.4_{\pm1.3}$ | $48.2_{\pm4.9}/36.6_{\pm4.6}$ | $75.8_{\pm2.2}$ | $54.5_{\pm3.7}/44.4_{\pm4.3}$ | $75.8_{\pm2.2}$ | $75.8_{\pm2.8}/82.9_{\pm2.0}$ | $75.8_{\pm2.2}$ | $43.6_{\pm2.7}/35.8_{\pm4.0}$ | $82.6_{\pm1.5}$ | $53.8_{\pm4.8}/42.0_{\pm5.6}$ | $91.2_{\pm1.6}$ |
| | GCNSafe | $90.5_{\pm0.8}/91.5_{\pm1.5}$ | $90.3_{\pm1.9}$ | $84.5_{\pm2.2}/85.1_{\pm2.9}$ | $87.4_{\pm1.3}$ | $48.2_{\pm4.9}/50.2_{\pm7.9}$ | $75.8_{\pm2.2}$ | $54.5_{\pm3.7}/54.8_{\pm4.4}$ | $75.8_{\pm2.2}$ | $75.8_{\pm2.8}/75.0_{\pm2.7}$ | $75.8_{\pm2.2}$ | $43.6_{\pm2.7}/39.9_{\pm4.8}$ | $82.6_{\pm1.5}$ | $53.8_{\pm4.8}/51.6_{\pm6.0}$ | $91.2_{\pm1.6}$ |
| | **GCN-GEBM** | $90.5_{\pm0.8}/91.5_{\pm1.4}$ | $90.3_{\pm1.9}$ | $84.5_{\pm2.2}/83.2_{\pm3.1}$ | $87.4_{\pm1.3}$ | $48.2_{\pm4.9}/47.1_{\pm8.2}$ | $75.8_{\pm2.2}$ | $54.5_{\pm3.7}/50.2_{\pm6.5}$ | $75.8_{\pm2.2}$ | $75.8_{\pm2.8}/82.2_{\pm3.6}$ | $75.8_{\pm2.2}$ | $43.6_{\pm2.7}/36.8_{\pm4.7}$ | $82.6_{\pm1.5}$ | $53.8_{\pm4.8}/46.9_{\pm4.8}$ | $91.2_{\pm1.6}$ |
| CoraML-LLM | GCN-MCD | $89.5_{\pm1.0}/51.8_{\pm2.5}$ | $90.1_{\pm1.0}$ | $84.1_{\pm4.8}/39.4_{\pm2.8}$ | $89.5_{\pm2.3}$ | n.a./n.a. | n.a. | $48.3_{\pm4.6}/25.5_{\pm3.2}$ | $76.9_{\pm1.4}$ | $65.2_{\pm1.7}/64.1_{\pm2.4}$ | $76.9_{\pm1.4}$ | $51.5_{\pm4.9}/21.4_{\pm3.5}$ | $84.0_{\pm1.5}$ | $54.5_{\pm5.2}/22.0_{\pm2.5}$ | $91.0_{\pm0.7}$ |
| | GCN-DE | $88.1_{\pm1.3}/70.9_{\pm2.9}$ | $87.8_{\pm1.2}$ | $82.4_{\pm3.7}/60.1_{\pm4.3}$ | $87.9_{\pm2.2}$ | n.a./n.a. | n.a. | $54.2_{\pm4.1}/31.9_{\pm3.7}$ | $75.6_{\pm1.5}$ | $66.3_{\pm2.7}/77.2_{\pm2.5}$ | $75.6_{\pm1.5}$ | $54.2_{\pm3.7}/39.5_{\pm3.8}$ | $81.8_{\pm2.2}$ | $52.6_{\pm4.9}/36.6_{\pm3.6}$ | $88.5_{\pm1.1}$ |
| | GCN-MCD+DE | $87.6_{\pm2.1}/71.0_{\pm3.3}$ | $88.1_{\pm1.4}$ | $81.7_{\pm4.2}/60.3_{\pm3.9}$ | $87.9_{\pm1.9}$ | n.a./n.a. | n.a. | $52.9_{\pm4.0}/32.3_{\pm4.2}$ | $75.2_{\pm2.1}$ | $66.3_{\pm2.9}/78.3_{\pm3.0}$ | $75.2_{\pm2.1}$ | $51.5_{\pm3.9}/37.6_{\pm2.5}$ | $81.3_{\pm1.8}$ | $52.4_{\pm5.5}/37.3_{\pm3.6}$ | $88.7_{\pm1.0}$ |
| | GCN-BNN | $89.3_{\pm1.5}/79.3_{\pm3.8}$ | $90.0_{\pm1.2}$ | $82.9_{\pm4.4}/75.8_{\pm4.6}$ | $89.3_{\pm2.2}$ | n.a./n.a. | n.a. | $46.7_{\pm3.5}/26.4_{\pm4.2}$ | $76.8_{\pm1.5}$ | $65.8_{\pm2.8}/74.9_{\pm2.3}$ | $76.8_{\pm1.5}$ | $53.6_{\pm6.0}/43.8_{\pm6.1}$ | $83.7_{\pm1.6}$ | $58.5_{\pm5.1}/49.6_{\pm4.7}$ | $90.8_{\pm0.9}$ |
| | GCN-Ens | $89.5_{\pm1.0}/77.5_{\pm4.6}$ | $90.1_{\pm1.0}$ | $84.1_{\pm4.3}/70.0_{\pm6.2}$ | $89.5_{\pm2.4}$ | n.a./n.a. | n.a. | $46.6_{\pm2.5}/21.4_{\pm3.2}$ | $77.3_{\pm1.3}$ | $63.8_{\pm3.0}/74.8_{\pm2.3}$ | $77.3_{\pm1.3}$ | $51.6_{\pm5.6}/34.1_{\pm3.5}$ | $84.2_{\pm1.2}$ | $53.8_{\pm4.8}/37.3_{\pm4.0}$ | $91.3_{\pm0.6}$ |
| | GPN | $85.6_{\pm2.6}/88.9_{\pm1.9}$ | $89.4_{\pm0.9}$ | $83.5_{\pm4.6}/75.7_{\pm3.8}$ | $88.1_{\pm2.4}$ | n.a./n.a. | n.a. | $53.1_{\pm4.6}/38.8_{\pm5.7}$ | $75.8_{\pm2.3}$ | $55.3_{\pm5.4}/41.1_{\pm4.8}$ | $75.8_{\pm2.3}$ | $54.4_{\pm5.7}/36.6_{\pm5.2}$ | $81.3_{\pm1.9}$ | $61.6_{\pm6.3}/44.8_{\pm5.6}$ | $90.8_{\pm1.2}$ |
| | SGCN | $90.0_{\pm1.3}/86.7_{\pm2.0}$ | $88.9_{\pm0.8}$ | $82.1_{\pm4.1}/77.4_{\pm7.2}$ | $89.8_{\pm1.9}$ | n.a./n.a. | n.a. | $59.7_{\pm12.2}/38.0_{\pm15.8}$ | $68.1_{\pm18.7}$ | $69.1_{\pm7.0}/66.7_{\pm8.0}$ | $68.1_{\pm18.7}$ | $54.6_{\pm10.6}/38.6_{\pm8.4}$ | $78.9_{\pm14.6}$ | $60.0_{\pm11.0}/46.4_{\pm12.0}$ | $80.2_{\pm20.4}$ |
| | GCN-EBM | $89.4_{\pm1.1}/88.7_{\pm1.4}$ | $90.2_{\pm0.9}$ | $83.8_{\pm4.5}/83.6_{\pm4.6}$ | $89.4_{\pm2.3}$ | n.a./n.a. | n.a. | $47.5_{\pm4.3}/42.9_{\pm3.4}$ | $77.1_{\pm1.3}$ | $67.2_{\pm3.0}/65.9_{\pm3.3}$ | $77.1_{\pm1.3}$ | $51.9_{\pm5.8}/44.5_{\pm4.6}$ | $83.7_{\pm1.5}$ | $54.6_{\pm3.9}/51.6_{\pm3.3}$ | $91.1_{\pm0.7}$ |
| | GCN-HEAT | $89.4_{\pm1.1}/86.5_{\pm3.0}$ | $90.2_{\pm0.9}$ | $83.8_{\pm4.5}/77.2_{\pm7.5}$ | $89.4_{\pm2.3}$ | n.a./n.a. | n.a. | $47.5_{\pm4.3}/36.2_{\pm4.9}$ | $77.1_{\pm1.3}$ | $67.2_{\pm3.0}/76.2_{\pm3.9}$ | $77.1_{\pm1.3}$ | $51.9_{\pm5.8}/31.9_{\pm4.2}$ | $83.7_{\pm1.5}$ | $54.6_{\pm3.9}/39.3_{\pm4.4}$ | $91.1_{\pm0.7}$ |
| | GCNSafe | $89.4_{\pm1.1}/90.6_{\pm1.5}$ | $90.2_{\pm0.9}$ | $83.8_{\pm4.5}/86.0_{\pm2.4}$ | $89.4_{\pm2.3}$ | n.a./n.a. | n.a. | $47.5_{\pm4.3}/51.7_{\pm4.7}$ | $77.1_{\pm1.3}$ | $67.2_{\pm3.0}/65.3_{\pm2.6}$ | $77.1_{\pm1.3}$ | $51.9_{\pm5.8}/43.8_{\pm2.8}$ | $83.7_{\pm1.5}$ | $54.6_{\pm3.9}/52.0_{\pm4.8}$ | $91.1_{\pm0.7}$ |
| | **GCN-GEBM** | $89.4_{\pm1.1}/89.1_{\pm1.5}$ | $90.2_{\pm0.9}$ | $83.8_{\pm4.5}/84.1_{\pm4.5}$ | $89.4_{\pm2.3}$ | n.a./n.a. | n.a. | $47.5_{\pm4.3}/37.8_{\pm4.7}$ | $77.1_{\pm1.3}$ | $67.2_{\pm3.0}/72.9_{\pm3.4}$ | $77.1_{\pm1.3}$ | $51.9_{\pm5.8}/44.1_{\pm7.3}$ | $83.7_{\pm1.5}$ | $54.6_{\pm3.9}/44.4_{\pm4.7}$ | $91.1_{\pm0.7}$ |
| Citeseer | GCN-MCD | $84.2_{\pm1.9}/41.7_{\pm1.8}$ | $84.8_{\pm1.2}$ | $86.2_{\pm2.3}/44.1_{\pm1.8}$ | $90.9_{\pm0.6}$ | $54.2_{\pm3.3}/36.3_{\pm2.3}$ | $76.4_{\pm1.5}$ | $63.4_{\pm3.4}/44.9_{\pm4.2}$ | $76.4_{\pm1.5}$ | $67.8_{\pm2.4}/71.3_{\pm3.1}$ | $76.4_{\pm1.5}$ | $50.2_{\pm3.6}/21.0_{\pm2.6}$ | $78.4_{\pm1.1}$ | $46.4_{\pm4.7}/20.2_{\pm3.6}$ | $77.8_{\pm1.9}$ |
| | GCN-DE | $78.8_{\pm3.5}/65.7_{\pm2.9}$ | $81.0_{\pm1.2}$ | $83.3_{\pm2.5}/66.9_{\pm3.3}$ | $86.4_{\pm1.0}$ | $56.3_{\pm3.3}/59.5_{\pm3.3}$ | $72.9_{\pm1.6}$ | $55.8_{\pm4.4}/73.9_{\pm2.5}$ | $72.9_{\pm1.6}$ | $63.9_{\pm1.8}/78.2_{\pm2.1}$ | $72.9_{\pm1.6}$ | $49.6_{\pm4.5}/42.8_{\pm3.5}$ | $75.6_{\pm1.4}$ | $48.8_{\pm4.2}/41.6_{\pm3.7}$ | $74.3_{\pm1.8}$ |
| | GCN-MCD+DE | $79.3_{\pm2.9}/66.5_{\pm3.2}$ | $81.3_{\pm1.2}$ | $83.9_{\pm1.5}/67.4_{\pm1.9}$ | $87.1_{\pm1.1}$ | $54.6_{\pm3.2}/59.4_{\pm2.5}$ | $73.0_{\pm1.5}$ | $55.2_{\pm4.3}/72.3_{\pm3.7}$ | $73.0_{\pm1.5}$ | $64.7_{\pm2.4}/79.0_{\pm2.1}$ | $73.0_{\pm1.5}$ | $49.4_{\pm4.4}/41.8_{\pm5.2}$ | $75.0_{\pm1.8}$ | $49.3_{\pm4.1}/42.2_{\pm3.6}$ | $74.6_{\pm1.6}$ |
| | GCN-BNN | $82.9_{\pm2.9}/82.7_{\pm1.8}$ | $84.5_{\pm1.3}$ | $85.4_{\pm2.2}/83.7_{\pm2.3}$ | $90.3_{\pm1.0}$ | $57.9_{\pm5.2}/56.4_{\pm7.1}$ | $75.8_{\pm1.5}$ | $64.3_{\pm4.0}/74.3_{\pm4.1}$ | $75.8_{\pm1.5}$ | $69.2_{\pm2.0}/78.5_{\pm2.4}$ | $75.8_{\pm1.5}$ | $51.8_{\pm4.0}/56.5_{\pm4.7}$ | $77.8_{\pm0.6}$ | $49.7_{\pm5.2}/53.6_{\pm3.2}$ | $77.1_{\pm2.0}$ |
| | GCN-Ens | $84.4_{\pm1.4}/71.5_{\pm4.5}$ | $84.8_{\pm1.0}$ | $85.9_{\pm2.4}/72.8_{\pm3.1}$ | $91.1_{\pm0.5}$ | $54.5_{\pm2.1}/39.2_{\pm3.4}$ | $76.9_{\pm1.6}$ | $59.1_{\pm1.7}/62.6_{\pm5.6}$ | $76.9_{\pm1.6}$ | $68.0_{\pm2.1}/77.1_{\pm1.4}$ | $76.9_{\pm1.6}$ | $49.2_{\pm3.1}/41.7_{\pm3.7}$ | $78.6_{\pm1.0}$ | $46.2_{\pm5.1}/37.4_{\pm4.0}$ | $77.7_{\pm1.8}$ |
| | GPN | $84.7_{\pm3.1}/84.0_{\pm2.7}$ | $85.6_{\pm1.6}$ | $88.8_{\pm2.2}/80.0_{\pm4.0}$ | $91.6_{\pm1.2}$ | $57.8_{\pm5.0}/53.0_{\pm5.6}$ | $76.7_{\pm1.9}$ | $61.7_{\pm4.5}/64.7_{\pm7.0}$ | $76.7_{\pm1.9}$ | $64.5_{\pm5.2}/56.0_{\pm4.2}$ | $76.7_{\pm1.9}$ | $57.2_{\pm5.1}/49.5_{\pm6.6}$ | $78.3_{\pm2.3}$ | $52.9_{\pm6.6}/45.8_{\pm7.1}$ | $77.4_{\pm2.2}$ |
| | SGCN | $83.8_{\pm2.3}/79.7_{\pm3.9}$ | $85.0_{\pm1.1}$ | $86.5_{\pm2.5}/80.7_{\pm4.0}$ | $90.4_{\pm0.7}$ | $58.8_{\pm3.9}/54.0_{\pm3.7}$ | $75.8_{\pm1.4}$ | $65.4_{\pm3.0}/59.8_{\pm4.6}$ | $75.8_{\pm1.4}$ | $69.6_{\pm2.2}/68.5_{\pm2.6}$ | $75.8_{\pm1.4}$ | $52.0_{\pm3.9}/48.4_{\pm4.4}$ | $79.3_{\pm1.1}$ | $51.0_{\pm5.2}/47.7_{\pm4.2}$ | $76.6_{\pm2.0}$ |
| | GCN-EBM | $84.3_{\pm1.4}/84.1_{\pm1.5}$ | $84.8_{\pm1.0}$ | $86.5_{\pm2.0}/85.5_{\pm2.5}$ | $90.9_{\pm0.7}$ | $56.6_{\pm3.2}/56.9_{\pm4.0}$ | $76.6_{\pm1.5}$ | $62.1_{\pm2.2}/54.4_{\pm4.7}$ | $76.6_{\pm1.5}$ | $68.8_{\pm2.0}/66.8_{\pm2.3}$ | $76.6_{\pm1.5}$ | $49.5_{\pm4.0}/48.6_{\pm5.0}$ | $78.7_{\pm1.1}$ | $46.7_{\pm4.7}/47.8_{\pm5.3}$ | $77.5_{\pm1.9}$ |
| | GCN-HEAT | $84.3_{\pm1.4}/77.9_{\pm2.7}$ | $84.8_{\pm1.0}$ | $86.5_{\pm2.0}/79.8_{\pm3.3}$ | $90.9_{\pm0.7}$ | $56.6_{\pm3.2}/46.7_{\pm3.6}$ | $76.6_{\pm1.5}$ | $62.1_{\pm2.2}/60.7_{\pm5.0}$ | $76.6_{\pm1.5}$ | $68.8_{\pm2.0}/80.3_{\pm2.9}$ | $76.6_{\pm1.5}$ | $49.5_{\pm4.0}/39.7_{\pm3.4}$ | $78.7_{\pm1.1}$ | $46.7_{\pm4.7}/39.1_{\pm4.8}$ | $77.5_{\pm1.9}$ |
| | GCNSafe | $84.3_{\pm1.4}/86.6_{\pm1.7}$ | $84.8_{\pm1.0}$ | $86.5_{\pm2.0}/87.4_{\pm3.1}$ | $90.9_{\pm0.7}$ | $56.6_{\pm3.2}/64.0_{\pm4.1}$ | $76.6_{\pm1.5}$ | $62.1_{\pm2.2}/62.2_{\pm4.0}$ | $76.6_{\pm1.5}$ | $68.8_{\pm2.0}/66.2_{\pm1.4}$ | $76.6_{\pm1.5}$ | $49.5_{\pm4.0}/56.2_{\pm5.5}$ | $78.7_{\pm1.1}$ | $46.7_{\pm4.7}/55.1_{\pm4.2}$ | $77.5_{\pm1.9}$ |
| | **GCN-GEBM** | $84.3_{\pm1.4}/85.4_{\pm1.8}$ | $84.8_{\pm1.0}$ | $86.5_{\pm2.0}/85.8_{\pm2.2}$ | $90.9_{\pm0.7}$ | $56.6_{\pm3.2}/55.1_{\pm2.9}$ | $76.6_{\pm1.5}$ | $62.1_{\pm2.2}/66.7_{\pm4.8}$ | $76.6_{\pm1.5}$ | $68.8_{\pm2.0}/76.7_{\pm2.0}$ | $76.6_{\pm1.5}$ | $49.5_{\pm4.0}/49.3_{\pm4.0}$ | $78.7_{\pm1.1}$ | $46.7_{\pm4.7}/49.4_{\pm3.9}$ | $77.5_{\pm1.9}$ |
| PubMed | GCN-MCD | $65.2_{\pm2.4}/43.4_{\pm0.9}$ | $92.3_{\pm1.8}$ | $57.7_{\pm1.9}/33.1_{\pm1.1}$ | $82.2_{\pm0.8}$ | n.a./n.a. | n.a. | $51.2_{\pm3.5}/34.3_{\pm3.4}$ | $76.9_{\pm1.5}$ | $57.4_{\pm1.1}/56.7_{\pm1.1}$ | $76.9_{\pm1.5}$ | $40.8_{\pm2.4}/21.4_{\pm1.1}$ | $78.7_{\pm0.9}$ | $41.9_{\pm2.0}/22.4_{\pm1.2}$ | $83.3_{\pm1.9}$ |
| | GCN-DE | $64.7_{\pm2.6}/54.8_{\pm1.7}$ | $91.8_{\pm1.9}$ | $56.4_{\pm2.2}/47.1_{\pm1.8}$ | $79.2_{\pm2.0}$ | n.a./n.a. | n.a. | $49.0_{\pm2.8}/51.9_{\pm1.5}$ | $74.9_{\pm1.7}$ | $55.2_{\pm1.6}/60.5_{\pm1.5}$ | $74.9_{\pm1.7}$ | $41.0_{\pm2.2}/33.6_{\pm1.9}$ | $76.8_{\pm1.6}$ | $42.2_{\pm3.1}/34.6_{\pm2.7}$ | $80.9_{\pm2.3}$ |
| | GCN-MCD+DE | $64.5_{\pm2.3}/54.8_{\pm1.3}$ | $91.7_{\pm1.9}$ | $56.8_{\pm2.8}/47.1_{\pm2.4}$ | $79.3_{\pm1.6}$ | n.a./n.a. | n.a. | $48.4_{\pm4.5}/51.5_{\pm1.7}$ | $74.9_{\pm1.6}$ | $55.0_{\pm1.3}/60.4_{\pm1.2}$ | $74.9_{\pm1.6}$ | $41.0_{\pm1.7}/33.4_{\pm1.6}$ | $77.1_{\pm1.3}$ | $42.6_{\pm2.4}/35.4_{\pm2.3}$ | $80.7_{\pm2.4}$ |
| | GCN-BNN | $64.5_{\pm3.1}/62.7_{\pm3.6}$ | $92.2_{\pm1.7}$ | $57.6_{\pm2.4}/55.3_{\pm3.9}$ | $81.7_{\pm3.1}$ | n.a./n.a. | n.a. | $53.6_{\pm4.9}/47.3_{\pm7.8}$ | $76.4_{\pm1.1}$ | $57.5_{\pm1.5}/61.9_{\pm1.5}$ | $76.4_{\pm1.1}$ | $42.3_{\pm4.0}/41.0_{\pm4.0}$ | $77.1_{\pm3.3}$ | $44.4_{\pm2.2}/43.2_{\pm3.4}$ | $82.8_{\pm2.2}$ |
| | GCN-Ens | $65.4_{\pm2.4}/50.2_{\pm3.6}$ | $92.3_{\pm1.8}$ | $57.9_{\pm2.1}/37.2_{\pm4.6}$ | $82.2_{\pm0.8}$ | n.a./n.a. | n.a. | $50.4_{\pm3.7}/41.8_{\pm7.5}$ | $76.9_{\pm1.5}$ | $57.1_{\pm1.4}/60.7_{\pm1.3}$ | $76.9_{\pm1.5}$ | $40.9_{\pm2.1}/27.1_{\pm3.5}$ | $78.7_{\pm0.8}$ | $41.7_{\pm2.0}/27.8_{\pm3.2}$ | $83.3_{\pm1.8}$ |
| | GPN | $62.5_{\pm3.3}/61.5_{\pm2.7}$ | $92.9_{\pm0.8}$ | $57.3_{\pm4.0}/55.2_{\pm4.8}$ | $85.0_{\pm1.3}$ | n.a./n.a. | n.a. | $49.9_{\pm4.3}/37.0_{\pm4.7}$ | $76.9_{\pm1.1}$ | $43.3_{\pm1.6}/30.1_{\pm1.9}$ | $76.9_{\pm1.1}$ | $42.6_{\pm2.1}/30.2_{\pm2.5}$ | $80.2_{\pm1.1}$ | $44.8_{\pm2.5}/32.7_{\pm2.1}$ | $86.2_{\pm1.5}$ |
| | SGCN | $64.3_{\pm1.9}/65.5_{\pm1.8}$ | $92.7_{\pm1.1}$ | $56.5_{\pm2.6}/53.6_{\pm3.3}$ | $83.8_{\pm1.2}$ | n.a./n.a. | n.a. | $50.6_{\pm3.5}/42.9_{\pm3.8}$ | $77.2_{\pm1.4}$ | $56.7_{\pm1.2}/56.2_{\pm1.4}$ | $77.2_{\pm1.4}$ | $41.3_{\pm2.4}/34.0_{\pm2.9}$ | $80.8_{\pm1.0}$ | $43.4_{\pm2.5}/37.3_{\pm4.5}$ | $83.7_{\pm1.9}$ |
| | GCN-EBM | $65.2_{\pm2.3}/64.7_{\pm2.3}$ | $92.3_{\pm1.8}$ | $58.0_{\pm1.8}/56.9_{\pm2.2}$ | $82.2_{\pm0.8}$ | n.a./n.a. | n.a. | $50.3_{\pm3.5}/42.9_{\pm4.0}$ | $76.9_{\pm1.4}$ | $56.4_{\pm1.0}/56.6_{\pm1.2}$ | $76.9_{\pm1.4}$ | $41.1_{\pm2.4}/38.7_{\pm2.6}$ | $78.6_{\pm0.9}$ | $41.7_{\pm2.4}/39.4_{\pm2.5}$ | $83.3_{\pm2.0}$ |
| | GCN-HEAT | $65.2_{\pm2.3}/61.3_{\pm3.3}$ | $92.3_{\pm1.8}$ | $58.0_{\pm1.8}/48.9_{\pm4.7}$ | $82.2_{\pm0.8}$ | n.a./n.a. | n.a. | $50.3_{\pm3.5}/40.5_{\pm4.5}$ | $76.9_{\pm1.4}$ | $56.4_{\pm1.0}/62.3_{\pm1.9}$ | $76.9_{\pm1.4}$ | $41.1_{\pm2.4}/31.4_{\pm2.9}$ | $78.6_{\pm0.9}$ | $41.7_{\pm2.4}/35.9_{\pm3.5}$ | $83.3_{\pm2.0}$ |
| | GCNSafe | $65.2_{\pm2.3}/64.2_{\pm3.9}$ | $92.3_{\pm1.8}$ | $58.0_{\pm1.8}/61.1_{\pm2.9}$ | $82.2_{\pm0.8}$ | n.a./n.a. | n.a. | $50.3_{\pm3.5}/46.9_{\pm3.9}$ | $76.9_{\pm1.4}$ | $56.4_{\pm1.0}/55.7_{\pm1.1}$ | $76.9_{\pm1.4}$ | $41.1_{\pm2.4}/37.4_{\pm2.0}$ | $78.6_{\pm0.9}$ | $41.7_{\pm2.4}/37.7_{\pm2.3}$ | $83.3_{\pm2.0}$ |
| | **GCN-GEBM** | $65.2_{\pm2.3}/65.0_{\pm1.7}$ | $92.3_{\pm1.8}$ | $58.0_{\pm1.8}/54.0_{\pm3.5}$ | $82.2_{\pm0.8}$ | n.a./n.a. | n.a. | $50.3_{\pm3.5}/44.1_{\pm5.8}$ | $76.9_{\pm1.4}$ | $56.4_{\pm1.0}/59.3_{\pm2.1}$ | $76.9_{\pm1.4}$ | $41.1_{\pm2.4}/31.9_{\pm3.8}$ | $78.6_{\pm0.9}$ | $41.7_{\pm2.4}/37.0_{\pm3.3}$ | $83.3_{\pm2.0}$ |
| Amazon Computers | GCN-MCD | $64.9_{\pm6.8}/37.7_{\pm1.7}$ | $88.8_{\pm1.8}$ | $85.1_{\pm4.8}/45.7_{\pm1.9}$ | $94.2_{\pm1.1}$ | $35.5_{\pm1.8}/22.7_{\pm1.7}$ | $80.2_{\pm2.5}$ | $51.2_{\pm13.4}/39.6_{\pm13.9}$ | $80.2_{\pm2.5}$ | $67.3_{\pm4.1}/67.0_{\pm2.6}$ | $80.2_{\pm2.5}$ | $39.8_{\pm4.7}/22.9_{\pm2.9}$ | $81.0_{\pm1.5}$ | $39.1_{\pm2.1}/22.4_{\pm1.6}$ | $91.2_{\pm1.3}$ |
| | GCN-DE | $66.2_{\pm6.0}/63.4_{\pm2.2}$ | $87.9_{\pm1.9}$ | $85.2_{\pm3.5}/72.1_{\pm2.9}$ | $93.4_{\pm1.3}$ | $35.5_{\pm2.7}/45.2_{\pm1.8}$ | $78.8_{\pm2.5}$ | $48.9_{\pm13.6}/53.2_{\pm4.5}$ | $78.8_{\pm2.5}$ | $71.6_{\pm4.2}/82.2_{\pm1.6}$ | $78.8_{\pm2.5}$ | $39.5_{\pm4.1}/44.3_{\pm2.2}$ | $80.4_{\pm1.3}$ | $38.0_{\pm2.0}/37.8_{\pm1.6}$ | $89.5_{\pm1.4}$ |
| | GCN-MCD+DE | $65.9_{\pm5.4}/63.3_{\pm1.6}$ | $88.2_{\pm1.7}$ | $84.7_{\pm4.0}/72.0_{\pm2.9}$ | $93.4_{\pm1.4}$ | $36.4_{\pm2.6}/45.0_{\pm2.5}$ | $78.5_{\pm2.8}$ | $48.6_{\pm14.3}/53.3_{\pm4.8}$ | $78.5_{\pm2.8}$ | $70.3_{\pm4.7}/81.3_{\pm1.3}$ | $78.5_{\pm2.8}$ | $39.5_{\pm3.9}/44.2_{\pm2.4}$ | $80.5_{\pm1.2}$ | $38.1_{\pm2.4}/37.7_{\pm1.5}$ | $89.8_{\pm1.4}$ |
| | GCN-BNN | $69.4_{\pm8.8}/73.9_{\pm9.2}$ | $84.3_{\pm9.1}$ | $83.7_{\pm4.7}/84.7_{\pm5.7}$ | $93.4_{\pm1.9}$ | $42.7_{\pm6.2}/42.7_{\pm5.0}$ | $78.3_{\pm2.5}$ | $57.4_{\pm14.0}/56.6_{\pm13.6}$ | $78.3_{\pm2.5}$ | $69.2_{\pm6.3}/79.6_{\pm3.1}$ | $78.3_{\pm2.5}$ | $39.5_{\pm4.8}/46.5_{\pm3.8}$ | $77.7_{\pm3.2}$ | $42.9_{\pm2.7}/48.6_{\pm3.2}$ | $90.8_{\pm1.4}$ |
| | GCN-Ens | $64.9_{\pm7.0}/66.9_{\pm6.0}$ | $88.9_{\pm1.9}$ | $85.0_{\pm4.9}/78.4_{\pm5.1}$ | $94.2_{\pm1.1}$ | $35.7_{\pm1.3}/30.8_{\pm3.1}$ | $80.4_{\pm2.5}$ | $50.8_{\pm13.7}/43.0_{\pm14.2}$ | $80.4_{\pm2.5}$ | $64.3_{\pm3.4}/75.4_{\pm1.5}$ | $80.4_{\pm2.5}$ | $39.1_{\pm4.6}/39.1_{\pm4.8}$ | $81.2_{\pm1.3}$ | $38.6_{\pm1.8}/41.8_{\pm3.5}$ | $91.3_{\pm1.2}$ |
| | GPN | $73.7_{\pm5.6}/65.0_{\pm5.2}$ | $86.4_{\pm1.9}$ | $76.8_{\pm7.9}/77.2_{\pm3.7}$ | $94.3_{\pm1.1}$ | $49.2_{\pm10.2}/39.0_{\pm11.3}$ | $74.4_{\pm14.7}$ | $48.7_{\pm9.9}/38.7_{\pm11.5}$ | $74.4_{\pm14.7}$ | $53.0_{\pm8.5}/41.4_{\pm10.0}$ | $74.4_{\pm14.7}$ | $46.5_{\pm4.1}/39.5_{\pm2.7}$ | $78.5_{\pm2.2}$ | $60.0_{\pm7.1}/45.9_{\pm5.8}$ | $92.4_{\pm1.2}$ |
| | SGCN | $73.1_{\pm16.2}/79.1_{\pm12.9}$ | $39.5_{\pm27.2}$ | $72.1_{\pm11.8}/80.3_{\pm10.0}$ | $59.1_{\pm26.4}$ | $88.4_{\pm14.4}/88.9_{\pm10.6}$ | $11.7_{\pm12.2}$ | $88.4_{\pm14.3}/88.9_{\pm10.6}$ | $11.7_{\pm12.2}$ | $87.9_{\pm11.8}/89.1_{\pm8.1}$ | $11.7_{\pm12.2}$ | $75.4_{\pm16.0}/83.1_{\pm10.8}$ | $20.0_{\pm14.3}$ | $92.2_{\pm10.2}/90.6_{\pm8.0}$ | $10.2_{\pm9.5}$ |
| | GCN-EBM | $65.6_{\pm6.6}/66.0_{\pm6.3}$ | $88.8_{\pm1.9}$ | $84.7_{\pm5.0}/82.4_{\pm4.5}$ | $94.1_{\pm1.2}$ | $36.4_{\pm1.8}/35.0_{\pm1.9}$ | $80.0_{\pm2.6}$ | $51.0_{\pm14.5}/50.0_{\pm14.6}$ | $80.0_{\pm2.6}$ | $69.6_{\pm3.8}/70.6_{\pm3.8}$ | $80.0_{\pm2.6}$ | $40.1_{\pm4.7}/36.5_{\pm4.2}$ | $81.0_{\pm1.7}$ | $38.8_{\pm2.2}/36.8_{\pm2.5}$ | $91.1_{\pm1.2}$ |
| | GCN-HEAT | $65.6_{\pm6.6}/61.4_{\pm7.0}$ | $88.8_{\pm1.9}$ | $84.7_{\pm5.0}/76.8_{\pm5.7}$ | $94.1_{\pm1.2}$ | $36.4_{\pm1.8}/31.4_{\pm2.7}$ | $80.0_{\pm2.6}$ | $51.0_{\pm14.5}/49.6_{\pm15.5}$ | $80.0_{\pm2.6}$ | $69.6_{\pm3.8}/76.4_{\pm2.2}$ | $80.0_{\pm2.6}$ | $40.1_{\pm4.7}/31.8_{\pm3.8}$ | $81.0_{\pm1.7}$ | $38.8_{\pm2.2}/33.8_{\pm2.2}$ | $91.1_{\pm1.2}$ |
| | GCNSafe | $65.6_{\pm6.6}/78.2_{\pm3.7}$ | $88.8_{\pm1.9}$ | $84.7_{\pm5.0}/81.0_{\pm4.7}$ | $94.1_{\pm1.2}$ | $36.4_{\pm1.8}/41.2_{\pm2.3}$ | $80.0_{\pm2.6}$ | $51.0_{\pm14.5}/55.7_{\pm12.4}$ | $80.0_{\pm2.6}$ | $69.6_{\pm3.8}/71.3_{\pm3.1}$ | $80.0_{\pm2.6}$ | $40.1_{\pm4.7}/45.1_{\pm3.9}$ | $81.0_{\pm1.7}$ | $38.8_{\pm2.2}/39.3_{\pm2.7}$ | $91.1_{\pm1.2}$ |
| | **GCN-GEBM** | $65.6_{\pm6.6}/75.6_{\pm4.3}$ | $88.8_{\pm1.9}$ | $84.7_{\pm5.0}/80.5_{\pm6.3}$ | $94.1_{\pm1.2}$ | $36.4_{\pm1.8}/31.4_{\pm1.9}$ | $80.0_{\pm2.6}$ | $51.0_{\pm14.5}/44.3_{\pm16.2}$ | $80.0_{\pm2.6}$ | $69.6_{\pm3.8}/70.8_{\pm2.7}$ | $80.0_{\pm2.6}$ | $40.1_{\pm4.7}/36.2_{\pm5.0}$ | $81.0_{\pm1.7}$ | $38.8_{\pm2.2}/34.6_{\pm3.5}$ | $91.1_{\pm1.2}$ |
| Amazon Photo | GCN-MCD | $73.5_{\pm4.2}/46.5_{\pm1.3}$ | $92.0_{\pm1.1}$ | $64.2_{\pm3.6}/32.3_{\pm2.2}$ | $95.3_{\pm2.0}$ | $36.5_{\pm4.9}/16.6_{\pm3.0}$ | $89.8_{\pm1.1}$ | $54.2_{\pm7.2}/41.8_{\pm8.0}$ | $89.8_{\pm1.1}$ | $68.2_{\pm3.0}/66.5_{\pm3.2}$ | $89.8_{\pm1.1}$ | $41.4_{\pm5.2}/13.5_{\pm2.2}$ | $91.9_{\pm0.5}$ | $36.7_{\pm3.3}/12.4_{\pm2.0}$ | $97.4_{\pm0.2}$ |
| | GCN-DE | $75.6_{\pm3.3}/57.6_{\pm2.3}$ | $91.7_{\pm1.1}$ | $66.7_{\pm3.9}/50.7_{\pm2.6}$ | $94.6_{\pm2.0}$ | $33.4_{\pm4.7}/39.3_{\pm3.1}$ | $88.5_{\pm1.2}$ | $53.3_{\pm6.9}/41.6_{\pm5.1}$ | $88.5_{\pm1.2}$ | $67.9_{\pm3.7}/78.1_{\pm2.1}$ | $88.5_{\pm1.2}$ | $40.7_{\pm5.8}/29.2_{\pm3.6}$ | $91.3_{\pm0.7}$ | $32.7_{\pm3.2}/22.1_{\pm2.3}$ | $96.6_{\pm0.4}$ |
| | GCN-MCD+DE | $75.8_{\pm3.3}/58.0_{\pm2.0}$ | $91.4_{\pm1.1}$ | $66.0_{\pm4.3}/49.9_{\pm2.3}$ | $94.8_{\pm1.9}$ | $34.6_{\pm5.0}/37.7_{\pm4.2}$ | $88.3_{\pm1.8}$ | $53.3_{\pm8.2}/42.1_{\pm5.5}$ | $88.3_{\pm1.8}$ | $69.3_{\pm3.6}/76.7_{\pm2.1}$ | $88.3_{\pm1.8}$ | $39.6_{\pm4.8}/29.2_{\pm4.3}$ | $91.6_{\pm0.7}$ | $32.7_{\pm3.8}/22.1_{\pm2.3}$ | $96.6_{\pm0.4}$ |
| | GCN-BNN | $71.7_{\pm5.4}/73.9_{\pm5.7}$ | $91.6_{\pm2.0}$ | $72.3_{\pm5.9}/79.0_{\pm6.7}$ | $95.7_{\pm1.9}$ | $46.3_{\pm8.9}/35.4_{\pm11.5}$ | $89.3_{\pm2.1}$ | $66.2_{\pm12.1}/62.2_{\pm12.0}$ | $89.3_{\pm2.1}$ | $66.5_{\pm2.7}/75.4_{\pm2.9}$ | $89.3_{\pm2.1}$ | $40.8_{\pm4.4}/31.0_{\pm3.8}$ | $90.4_{\pm2.7}$ | $39.3_{\pm3.4}/41.2_{\pm6.2}$ | $97.6_{\pm0.4}$ |
| | GCN-Ens | $73.8_{\pm4.2}/77.3_{\pm8.1}$ | $91.9_{\pm1.0}$ | $64.1_{\pm3.8}/69.3_{\pm9.1}$ | $95.3_{\pm1.9}$ | $33.5_{\pm3.1}/18.2_{\pm3.8}$ | $89.9_{\pm1.0}$ | $53.7_{\pm7.1}/41.9_{\pm8.5}$ | $89.9_{\pm1.0}$ | $66.1_{\pm4.2}/73.6_{\pm3.6}$ | $89.9_{\pm1.0}$ | $41.9_{\pm5.6}/24.8_{\pm5.1}$ | $92.1_{\pm0.5}$ | $36.6_{\pm3.3}/26.8_{\pm2.2}$ | $97.5_{\pm0.2}$ |
| | GPN | $73.9_{\pm4.0}/85.6_{\pm2.9}$ | $88.8_{\pm1.3}$ | $79.8_{\pm5.5}/56.9_{\pm5.8}$ | $93.9_{\pm2.2}$ | $42.4_{\pm6.4}/28.9_{\pm5.9}$ | $86.5_{\pm1.8}$ | $41.6_{\pm6.7}/29.1_{\pm6.9}$ | $86.5_{\pm1.8}$ | $45.9_{\pm5.9}/29.7_{\pm4.5}$ | $86.5_{\pm1.8}$ | $43.7_{\pm7.0}/23.6_{\pm4.4}$ | $88.6_{\pm2.0}$ | $53.3_{\pm7.6}/26.6_{\pm6.8}$ | $97.1_{\pm0.5}$ |
| | SGCN | $74.3_{\pm3.2}/73.4_{\pm5.2}$ | $92.1_{\pm0.9}$ | $62.8_{\pm5.9}/61.8_{\pm9.1}$ | $94.3_{\pm2.2}$ | $85.5_{\pm10.6}/86.2_{\pm9.3}$ | $15.4_{\pm10.0}$ | $86.3_{\pm9.6}/86.7_{\pm8.4}$ | $15.4_{\pm10.0}$ | $83.5_{\pm9.1}/86.3_{\pm6.5}$ | $15.4_{\pm10.0}$ | $89.7_{\pm7.3}/88.6_{\pm9.6}$ | $12.2_{\pm8.6}$ | $87.4_{\pm8.8}/86.2_{\pm6.5}$ | $15.0_{\pm5.9}$ |
| | GCN-EBM | $73.8_{\pm4.4}/73.7_{\pm5.5}$ | $91.8_{\pm1.0}$ | $64.2_{\pm3.9}/58.9_{\pm5.0}$ | $95.3_{\pm2.0}$ | $35.7_{\pm5.9}/29.6_{\pm6.0}$ | $89.8_{\pm1.0}$ | $53.6_{\pm6.7}/52.2_{\pm5.9}$ | $89.8_{\pm1.0}$ | $66.0_{\pm3.7}/64.2_{\pm4.2}$ | $89.8_{\pm1.0}$ | $42.2_{\pm5.7}/35.9_{\pm5.8}$ | $91.9_{\pm0.6}$ | $35.9_{\pm4.3}/34.0_{\pm5.0}$ | $97.4_{\pm0.3}$ |
| | GCN-HEAT | $73.8_{\pm4.4}/70.9_{\pm4.1}$ | $91.8_{\pm1.0}$ | $64.2_{\pm3.9}/52.1_{\pm5.2}$ | $95.3_{\pm2.0}$ | $35.7_{\pm5.9}/24.2_{\pm5.1}$ | $89.8_{\pm1.0}$ | $53.6_{\pm6.7}/51.2_{\pm6.9}$ | $89.8_{\pm1.0}$ | $66.0_{\pm3.7}/72.1_{\pm2.4}$ | $89.8_{\pm1.0}$ | $42.2_{\pm5.7}/23.9_{\pm4.5}$ | $91.9_{\pm0.6}$ | $35.9_{\pm4.3}/24.2_{\pm2.4}$ | $97.4_{\pm0.3}$ |
| | GCNSafe | $73.8_{\pm4.4}/69.8_{\pm7.2}$ | $91.8_{\pm1.0}$ | $64.2_{\pm3.9}/66.2_{\pm6.6}$ | $95.3_{\pm2.0}$ | $35.7_{\pm5.9}/29.1_{\pm3.9}$ | $89.8_{\pm1.0}$ | $53.6_{\pm6.7}/51.2_{\pm5.8}$ | $89.8_{\pm1.0}$ | $66.0_{\pm3.7}/67.2_{\pm3.3}$ | $89.8_{\pm1.0}$ | $42.2_{\pm5.7}/28.5_{\pm5.9}$ | $91.9_{\pm0.6}$ | $35.9_{\pm4.3}/24.5_{\pm2.7}$ | $97.4_{\pm0.3}$ |
| | **GCN-GEBM** | $73.8_{\pm4.4}/81.3_{\pm6.5}$ | $91.8_{\pm1.0}$ | $64.2_{\pm3.9}/69.8_{\pm5.1}$ | $95.3_{\pm2.0}$ | $35.7_{\pm5.9}/15.7_{\pm1.9}$ | $89.8_{\pm1.0}$ | $53.6_{\pm6.7}/48.4_{\pm8.2}$ | $89.8_{\pm1.0}$ | $66.0_{\pm3.7}/68.3_{\pm3.9}$ | $89.8_{\pm1.0}$ | $42.2_{\pm5.7}/15.5_{\pm2.6}$ | $91.9_{\pm0.6}$ | $35.9_{\pm4.3}/23.7_{\pm3.6}$ | $97.4_{\pm0.3}$ |
| Coauthor-CS | GCN-MCD | $91.9_{\pm1.2}/60.5_{\pm1.2}$ | $92.3_{\pm0.8}$ | $81.3_{\pm2.1}/28.9_{\pm1.2}$ | $95.1_{\pm0.5}$ | $33.3_{\pm1.9}/13.3_{\pm1.3}$ | $89.9_{\pm1.0}$ | $46.0_{\pm4.4}/17.9_{\pm3.1}$ | $89.9_{\pm1.0}$ | $79.0_{\pm1.1}/79.6_{\pm1.5}$ | $89.9_{\pm1.0}$ | $32.4_{\pm1.7}/11.2_{\pm1.4}$ | $90.8_{\pm0.4}$ | $31.0_{\pm1.2}/10.7_{\pm1.2}$ | $97.9_{\pm0.4}$ |
| | GCN-DE | $91.8_{\pm0.8}/74.2_{\pm1.2}$ | $91.3_{\pm0.7}$ | $78.5_{\pm1.9}/48.6_{\pm2.0}$ | $94.4_{\pm0.7}$ | $32.4_{\pm2.2}/37.7_{\pm1.8}$ | $88.8_{\pm1.1}$ | $41.3_{\pm4.7}/53.5_{\pm3.4}$ | $88.8_{\pm1.1}$ | $76.7_{\pm1.1}/88.8_{\pm0.9}$ | $88.8_{\pm1.1}$ | $29.8_{\pm1.8}/22.9_{\pm2.3}$ | $89.6_{\pm0.6}$ | $28.1_{\pm1.7}/20.4_{\pm1.3}$ | $97.1_{\pm0.5}$ |
| | GCN-MCD+DE | $91.6_{\pm1.2}/74.0_{\pm1.5}$ | $91.4_{\pm0.5}$ | $77.4_{\pm2.1}/48.5_{\pm1.9}$ | $94.3_{\pm0.6}$ | $32.1_{\pm2.0}/37.6_{\pm2.1}$ | $88.5_{\pm1.0}$ | $39.5_{\pm5.0}/53.1_{\pm3.3}$ | $88.5_{\pm1.0}$ | $78.2_{\pm1.3}/89.0_{\pm1.0}$ | $88.5_{\pm1.0}$ | $29.6_{\pm1.8}/23.1_{\pm1.9}$ | $89.6_{\pm0.6}$ | $27.8_{\pm2.1}/20.4_{\pm1.9}$ | $97.1_{\pm0.4}$ |
| | GCN-BNN | $88.9_{\pm1.6}/87.4_{\pm1.9}$ | $90.8_{\pm0.8}$ | $76.7_{\pm2.0}/70.6_{\pm4.1}$ | $94.7_{\pm0.5}$ | $38.1_{\pm2.2}/32.6_{\pm2.8}$ | $89.0_{\pm0.8}$ | $54.7_{\pm6.0}/70.2_{\pm7.8}$ | $89.0_{\pm0.8}$ | $84.1_{\pm1.6}/90.1_{\pm1.2}$ | $89.0_{\pm0.8}$ | $39.8_{\pm2.6}/37.5_{\pm3.3}$ | $89.8_{\pm0.6}$ | $38.0_{\pm2.3}/37.5_{\pm2.2}$ | $97.9_{\pm0.4}$ |
| | GCN-Ens | $91.9_{\pm1.2}/87.7_{\pm2.3}$ | $92.4_{\pm0.8}$ | $81.3_{\pm1.8}/68.0_{\pm2.7}$ | $95.2_{\pm0.5}$ | $31.9_{\pm1.7}/14.8_{\pm1.3}$ | $90.0_{\pm0.9}$ | $44.0_{\pm3.1}/20.6_{\pm1.9}$ | $90.0_{\pm0.9}$ | $77.0_{\pm1.1}/84.9_{\pm1.1}$ | $90.0_{\pm0.9}$ | $32.2_{\pm1.7}/24.6_{\pm2.1}$ | $90.8_{\pm0.4}$ | $30.3_{\pm1.2}/25.0_{\pm1.3}$ | $97.9_{\pm0.4}$ |
| | GPN | $81.8_{\pm2.6}/90.1_{\pm1.2}$ | $86.6_{\pm1.4}$ | $67.0_{\pm3.4}/51.1_{\pm5.4}$ | $91.6_{\pm1.2}$ | $45.8_{\pm3.8}/24.3_{\pm3.3}$ | $81.7_{\pm2.3}$ | $47.9_{\pm4.6}/24.5_{\pm3.5}$ | $81.7_{\pm2.3}$ | $47.9_{\pm4.6}/24.4_{\pm3.5}$ | $81.7_{\pm2.3}$ | $45.9_{\pm4.9}/22.9_{\pm2.4}$ | $83.4_{\pm1.6}$ | $64.0_{\pm5.8}/47.3_{\pm11.1}$ | $97.9_{\pm0.4}$ |
| | SGCN | $93.7_{\pm2.7}/82.2_{\pm11.3}$ | $30.6_{\pm14.7}$ | $86.4_{\pm6.8}/72.1_{\pm16.5}$ | $31.2_{\pm16.2}$ | $94.2_{\pm4.6}/92.9_{\pm6.2}$ | $11.0_{\pm7.0}$ | $93.6_{\pm5.4}/93.1_{\pm6.0}$ | $11.0_{\pm7.0}$ | $92.4_{\pm2.4}/92.6_{\pm3.6}$ | $11.0_{\pm7.0}$ | $91.3_{\pm5.4}/88.2_{\pm11.6}$ | $12.1_{\pm9.1}$ | $91.3_{\pm3.8}/89.8_{\pm8.6}$ | $14.1_{\pm6.5}$ |
| | GCN-EBM | $91.9_{\pm1.2}/91.6_{\pm1.4}$ | $92.3_{\pm0.9}$ | $81.1_{\pm2.0}/79.7_{\pm2.5}$ | $95.1_{\pm0.5}$ | $33.6_{\pm2.7}/32.3_{\pm3.1}$ | $89.8_{\pm0.9}$ | $46.7_{\pm4.8}/49.1_{\pm5.6}$ | $89.8_{\pm0.9}$ | $79.0_{\pm1.3}/76.6_{\pm1.3}$ | $89.8_{\pm0.9}$ | $32.4_{\pm1.2}/26.0_{\pm1.2}$ | $90.9_{\pm0.3}$ | $31.2_{\pm1.3}/27.1_{\pm1.5}$ | $97.8_{\pm0.4}$ |
| | GCN-HEAT | $91.9_{\pm1.2}/88.3_{\pm2.1}$ | $92.3_{\pm0.9}$ | $81.1_{\pm2.0}/63.6_{\pm5.0}$ | $95.1_{\pm0.5}$ | $33.6_{\pm2.7}/23.1_{\pm2.9}$ | $89.8_{\pm0.9}$ | $46.7_{\pm4.8}/30.8_{\pm4.7}$ | $89.8_{\pm0.9}$ | $79.0_{\pm1.3}/87.3_{\pm1.2}$ | $89.8_{\pm0.9}$ | $32.4_{\pm1.2}/20.5_{\pm2.9}$ | $90.9_{\pm0.3}$ | $31.2_{\pm1.3}/30.0_{\pm2.9}$ | $97.8_{\pm0.4}$ |
| | GCNSafe | $91.9_{\pm1.2}/93.4_{\pm1.2}$ | $92.3_{\pm0.9}$ | $81.1_{\pm2.0}/81.1_{\pm2.1}$ | $95.1_{\pm0.5}$ | $33.6_{\pm2.7}/29.4_{\pm3.0}$ | $89.8_{\pm0.9}$ | $46.7_{\pm4.8}/43.8_{\pm4.9}$ | $89.8_{\pm0.9}$ | $79.0_{\pm1.3}/77.1_{\pm1.3}$ | $89.8_{\pm0.9}$ | $32.4_{\pm1.2}/20.2_{\pm1.2}$ | $90.9_{\pm0.3}$ | $31.2_{\pm1.3}/19.5_{\pm1.2}$ | $97.8_{\pm0.4}$ |
| | **GCN-GEBM** | $91.9_{\pm1.2}/94.0_{\pm1.0}$ | $92.3_{\pm0.9}$ | $81.1_{\pm2.0}/79.7_{\pm3.6}$ | $95.1_{\pm0.5}$ | $33.6_{\pm2.7}/24.1_{\pm3.8}$ | $89.8_{\pm0.9}$ | $46.7_{\pm4.8}/37.9_{\pm6.9}$ | $89.8_{\pm0.9}$ | $79.0_{\pm1.3}/86.0_{\pm1.9}$ | $89.8_{\pm0.9}$ | $32.4_{\pm1.2}/18.8_{\pm2.1}$ | $90.9_{\pm0.3}$ | $31.2_{\pm1.3}/21.8_{\pm2.2}$ | $97.8_{\pm0.4}$ |
| Coauthor-Physics | GCN-MCD | $81.1_{\pm2.3}/21.0_{\pm0.8}$ | $96.7_{\pm0.5}$ | $91.7_{\pm1.4}/26.8_{\pm0.7}$ | $98.3_{\pm0.2}$ | $38.9_{\pm3.4}/11.1_{\pm1.9}$ | $91.9_{\pm1.2}$ | $33.7_{\pm2.9}/11.4_{\pm2.1}$ | $91.9_{\pm1.2}$ | $61.5_{\pm1.8}/57.5_{\pm3.2}$ | $91.9_{\pm1.2}$ | $34.6_{\pm2.6}/7.7_{\pm1.3}$ | $93.4_{\pm1.0}$ | $40.8_{\pm2.0}/7.4_{\pm0.6}$ | $98.0_{\pm0.2}$ |
| | GCN-DE | $81.2_{\pm2.7}/58.2_{\pm2.3}$ | $96.4_{\pm0.7}$ | $90.8_{\pm1.4}/74.7_{\pm1.5}$ | $98.1_{\pm0.2}$ | $39.8_{\pm4.2}/36.6_{\pm2.9}$ | $91.1_{\pm1.2}$ | $32.5_{\pm3.8}/34.2_{\pm3.5}$ | $91.1_{\pm1.2}$ | $63.9_{\pm3.5}/70.9_{\pm2.0}$ | $91.1_{\pm1.2}$ | $33.3_{\pm2.9}/25.5_{\pm2.2}$ | $93.1_{\pm1.0}$ | $37.7_{\pm1.9}/23.4_{\pm2.1}$ | $97.7_{\pm0.2}$ |
| | GCN-MCD+DE | $81.4_{\pm2.5}/58.7_{\pm2.1}$ | $96.5_{\pm0.6}$ | $91.2_{\pm1.4}/75.0_{\pm1.8}$ | $98.2_{\pm0.2}$ | $40.0_{\pm3.4}/37.2_{\pm2.9}$ | $91.3_{\pm1.5}$ | $31.7_{\pm4.9}/34.2_{\pm3.9}$ | $91.3_{\pm1.5}$ | $62.7_{\pm2.1}/70.6_{\pm1.1}$ | $91.3_{\pm1.5}$ | $34.0_{\pm2.6}/25.3_{\pm2.3}$ | $93.1_{\pm0.9}$ | $37.3_{\pm1.9}/23.7_{\pm1.7}$ | $97.7_{\pm0.2}$ |
| | GCN-BNN | $74.9_{\pm5.8}/69.8_{\pm8.4}$ | $96.6_{\pm1.1}$ | $85.9_{\pm4.0}/86.4_{\pm4.2}$ | $98.1_{\pm0.3}$ | $38.1_{\pm4.7}/33.4_{\pm5.7}$ | $91.7_{\pm1.4}$ | $43.5_{\pm9.5}/51.3_{\pm13.0}$ | $91.7_{\pm1.4}$ | $74.6_{\pm9.2}/77.7_{\pm3.9}$ | $91.7_{\pm1.4}$ | $36.3_{\pm2.7}/36.7_{\pm3.5}$ | $93.4_{\pm0.9}$ | $42.1_{\pm3.7}/38.3_{\pm2.8}$ | $98.4_{\pm0.2}$ |
| | GCN-Ens | $81.2_{\pm2.4}/66.9_{\pm5.2}$ | $96.8_{\pm0.5}$ | $91.8_{\pm1.3}/83.0_{\pm2.6}$ | $98.3_{\pm0.2}$ | $39.5_{\pm4.0}/15.8_{\pm3.5}$ | $92.0_{\pm1.2}$ | $33.9_{\pm2.3}/13.6_{\pm2.3}$ | $92.0_{\pm1.2}$ | $60.7_{\pm2.1}/64.7_{\pm1.8}$ | $92.0_{\pm1.2}$ | $34.4_{\pm2.6}/26.4_{\pm3.5}$ | $93.3_{\pm1.0}$ | $40.7_{\pm2.1}/28.0_{\pm1.1}$ | $98.0_{\pm0.2}$ |
| | GPN | $68.9_{\pm3.8}/75.2_{\pm3.9}$ | $96.8_{\pm0.4}$ | $65.1_{\pm4.8}/84.4_{\pm2.4}$ | $97.7_{\pm0.3}$ | $42.4_{\pm3.9}/20.7_{\pm4.5}$ | $90.3_{\pm1.4}$ | $40.6_{\pm3.9}/18.3_{\pm4.0}$ | $90.3_{\pm1.4}$ | $40.6_{\pm3.7}/18.2_{\pm3.9}$ | $90.3_{\pm1.4}$ | $39.4_{\pm2.9}/16.9_{\pm2.6}$ | $92.3_{\pm1.0}$ | $51.4_{\pm11.3}/38.5_{\pm17.7}$ | $98.8_{\pm0.2}$ |
| | SGCN | $80.5_{\pm2.6}/77.7_{\pm5.5}$ | $96.9_{\pm0.4}$ | $91.8_{\pm1.2}/89.3_{\pm3.4}$ | $98.3_{\pm0.2}$ | $42.8_{\pm4.3}/29.4_{\pm8.1}$ | $91.3_{\pm1.4}$ | $41.7_{\pm5.7}/24.4_{\pm9.4}$ | $91.3_{\pm1.4}$ | $61.2_{\pm2.4}/56.2_{\pm3.9}$ | $91.3_{\pm1.4}$ | $34.0_{\pm2.7}/21.7_{\pm4.1}$ | $93.3_{\pm0.7}$ | $39.9_{\pm2.7}/27.6_{\pm6.5}$ | $97.5_{\pm0.6}$ |
| | GCN-EBM | $81.2_{\pm2.4}/82.1_{\pm3.1}$ | $96.8_{\pm0.5}$ | $91.7_{\pm1.1}/92.4_{\pm1.2}$ | $98.3_{\pm0.2}$ | $38.9_{\pm3.8}/35.7_{\pm5.7}$ | $92.0_{\pm1.2}$ | $33.8_{\pm2.8}/30.0_{\pm3.5}$ | $92.0_{\pm1.2}$ | $60.5_{\pm2.5}/57.8_{\pm3.4}$ | $92.0_{\pm1.2}$ | $34.1_{\pm2.6}/29.0_{\pm2.8}$ | $93.4_{\pm1.0}$ | $40.6_{\pm1.9}/36.0_{\pm2.4}$ | $98.0_{\pm0.2}$ |
| | GCN-HEAT | $81.2_{\pm2.4}/68.1_{\pm3.4}$ | $96.8_{\pm0.5}$ | $91.7_{\pm1.1}/80.3_{\pm1.8}$ | $98.3_{\pm0.2}$ | $38.9_{\pm3.8}/25.4_{\pm5.2}$ | $92.0_{\pm1.2}$ | $33.8_{\pm2.8}/24.8_{\pm4.2}$ | $92.0_{\pm1.2}$ | $60.5_{\pm2.5}/63.7_{\pm2.1}$ | $92.0_{\pm1.2}$ | $34.1_{\pm2.6}/25.0_{\pm3.8}$ | $93.4_{\pm1.0}$ | $40.6_{\pm1.9}/29.3_{\pm2.0}$ | $98.0_{\pm0.2}$ |
| | GCNSafe | $81.2_{\pm2.4}/86.5_{\pm2.5}$ | $96.8_{\pm0.5}$ | $91.7_{\pm1.1}/94.2_{\pm0.7}$ | $98.3_{\pm0.2}$ | $38.9_{\pm3.8}/36.0_{\pm5.9}$ | $92.0_{\pm1.2}$ | $33.8_{\pm2.8}/30.9_{\pm4.5}$ | $92.0_{\pm1.2}$ | $60.5_{\pm2.5}/59.8_{\pm2.3}$ | $92.0_{\pm1.2}$ | $34.1_{\pm2.6}/26.6_{\pm4.0}$ | $93.4_{\pm1.0}$ | $40.6_{\pm1.9}/28.9_{\pm2.2}$ | $98.0_{\pm0.2}$ |
| | **GCN-GEBM** | $81.2_{\pm2.4}/86.6_{\pm3.0}$ | $96.8_{\pm0.5}$ | $91.7_{\pm1.1}/94.9_{\pm1.0}$ | $98.3_{\pm0.2}$ | $38.9_{\pm3.8}/31.4_{\pm7.9}$ | $92.0_{\pm1.2}$ | $33.8_{\pm2.8}/27.3_{\pm4.0}$ | $92.0_{\pm1.2}$ | $60.5_{\pm2.5}/59.6_{\pm4.1}$ | $92.0_{\pm1.2}$ | $34.1_{\pm2.6}/30.1_{\pm5.3}$ | $93.4_{\pm1.0}$ | $40.6_{\pm1.9}/35.8_{\pm3.3}$ | $98.0_{\pm0.2}$ |

Table 18: Misclassification detection AUC-PR (↑) using aleatoric or epistemic uncertainty (best and runner-up) in an inductive setting.

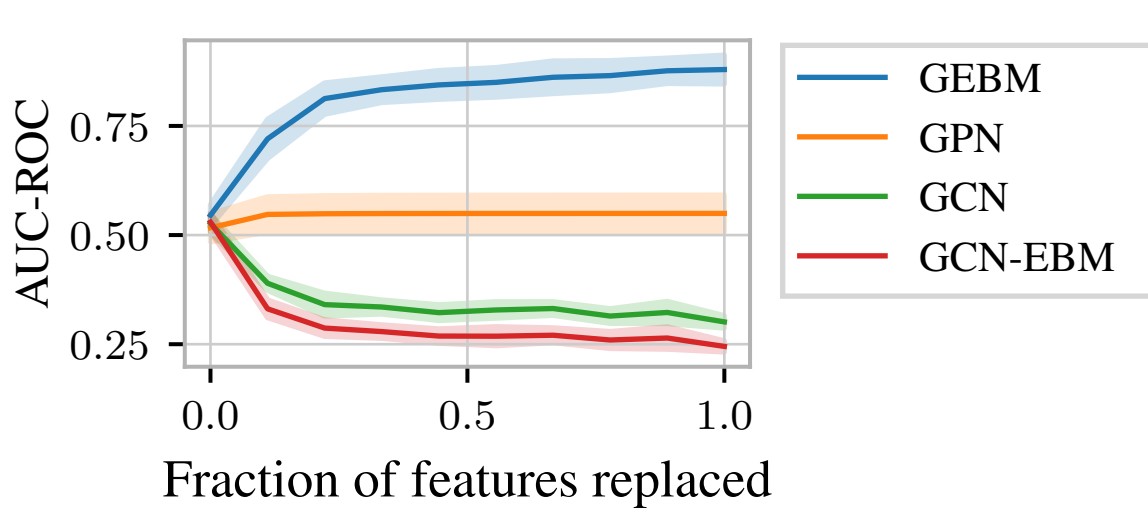

Figure 7: Out-of-distribution detection AUC-ROC for different estimators for an increasing fraction of perturbed node feature dimensions on CoraML.

| Dataset | Model | LoC *(last)* AUC-PR↑ (Alea. / Epi.) | Acc.↑ | LoC *(hetero.)* AUC-PR↑ (Alea. / Epi.) | Acc.↑ | Ber($\hat{p}$) *(near)* AUC-PR↑ (Alea. / Epi.) | Acc.↑ | Ber(0.5) AUC-PR↑ (Alea. / Epi.) | Acc.↑ | $\mathcal{N}(0,1)$ *(far)* AUC-PR↑ (Alea. / Epi.) | Acc.↑ | Page Rank AUC-PR↑ (Alea. / Epi.) | Acc.↑ | Homophily AUC-PR↑ (Alea. / Epi.) | Acc.↑ |
|---|---|---|---|---|---|---|---|---|---|---|---|---|---|---|---|
| CoraML | GCN-MCD | $88.6_{\pm0.8}/51.5_{\pm2.4}$ | $90.4_{\pm0.9}$ | $80.4_{\pm2.5}/43.3_{\pm3.7}$ | $85.2_{\pm3.3}$ | $53.8_{\pm3.7}/29.6_{\pm3.2}$ | $82.0_{\pm0.8}$ | $54.8_{\pm3.4}/32.0_{\pm3.9}$ | $82.0_{\pm0.8}$ | $76.6_{\pm2.0}/77.6_{\pm2.0}$ | $82.0_{\pm0.8}$ | $45.3_{\pm6.0}/22.6_{\pm4.0}$ | $82.6_{\pm1.5}$ | $52.8_{\pm3.4}/22.2_{\pm2.8}$ | $93.9_{\pm0.8}$ |
| | GCN-DE | $86.3_{\pm1.8}/71.6_{\pm2.9}$ | $87.3_{\pm1.2}$ | $78.7_{\pm2.8}/59.9_{\pm3.6}$ | $80.9_{\pm2.3}$ | $51.5_{\pm4.3}/54.7_{\pm4.9}$ | $78.4_{\pm1.9}$ | $52.6_{\pm4.9}/58.0_{\pm6.3}$ | $78.4_{\pm1.9}$ | $72.8_{\pm3.3}/81.0_{\pm2.6}$ | $78.4_{\pm1.9}$ | $46.5_{\pm3.5}/34.1_{\pm3.6}$ | $79.0_{\pm2.3}$ | $50.6_{\pm2.9}/36.4_{\pm4.0}$ | $90.6_{\pm1.8}$ |
| | GCN-MCD+DE | $86.7_{\pm1.7}/72.8_{\pm2.1}$ | $87.6_{\pm1.2}$ | $78.3_{\pm3.3}/61.2_{\pm3.6}$ | $80.1_{\pm2.9}$ | $51.8_{\pm4.1}/55.3_{\pm5.8}$ | $78.1_{\pm2.2}$ | $53.2_{\pm5.7}/59.7_{\pm5.8}$ | $78.1_{\pm2.2}$ | $70.4_{\pm2.5}/80.7_{\pm1.9}$ | $78.1_{\pm2.2}$ | $46.5_{\pm4.7}/37.4_{\pm4.3}$ | $79.5_{\pm1.7}$ | $51.8_{\pm3.8}/37.6_{\pm4.1}$ | $90.8_{\pm1.7}$ |
| | GCN-BNN | $86.0_{\pm2.4}/86.1_{\pm2.9}$ | $89.0_{\pm1.2}$ | $76.9_{\pm3.5}/73.7_{\pm4.1}$ | $82.1_{\pm3.4}$ | $55.7_{\pm5.1}/50.8_{\pm5.8}$ | $80.6_{\pm2.0}$ | $64.5_{\pm6.4}/64.7_{\pm10.5}$ | $80.6_{\pm2.0}$ | $81.0_{\pm3.0}/\mathbf{83.9}_{\pm2.7}$ | $80.6_{\pm2.0}$ | $48.6_{\pm7.3}/44.2_{\pm7.9}$ | $80.4_{\pm2.3}$ | $\mathbf{53.4}_{\pm5.2}/50.1_{\pm6.3}$ | $93.1_{\pm1.5}$ |
| | GCN-Ens | $88.8_{\pm0.6}/74.8_{\pm3.1}$ | $90.3_{\pm0.9}$ | $80.9_{\pm2.6}/53.3_{\pm5.2}$ | $85.2_{\pm3.3}$ | $53.1_{\pm3.7}/29.3_{\pm3.3}$ | $81.8_{\pm0.8}$ | $54.2_{\pm5.2}/34.4_{\pm5.5}$ | $81.8_{\pm0.8}$ | $76.7_{\pm3.6}/81.6_{\pm2.2}$ | $81.8_{\pm0.8}$ | $44.6_{\pm5.7}/27.8_{\pm5.7}$ | $82.7_{\pm1.5}$ | $\mathbf{53.4}_{\pm3.4}/33.3_{\pm3.1}$ | $94.0_{\pm0.5}$ |
| | GPN | $86.1_{\pm2.1}/87.7_{\pm1.9}$ | $90.0_{\pm1.0}$ | $77.7_{\pm5.9}/73.7_{\pm4.3}$ | $85.5_{\pm2.2}$ | $57.1_{\pm4.6}/41.8_{\pm7.4}$ | $78.8_{\pm2.1}$ | $56.3_{\pm5.4}/40.9_{\pm11.8}$ | $78.8_{\pm2.1}$ | $58.0_{\pm4.3}/40.4_{\pm4.5}$ | $78.8_{\pm2.1}$ | $\mathbf{55.5}_{\pm5.1}/40.5_{\pm6.1}$ | $78.4_{\pm1.8}$ | $\mathbf{55.6}_{\pm5.3}/41.9_{\pm7.2}$ | $91.1_{\pm1.8}$ |
| | SGCN | $88.7_{\pm1.4}/89.4_{\pm1.5}$ | $91.0_{\pm1.2}$ | $82.0_{\pm3.6}/\mathbf{83.2}_{\pm4.1}$ | $88.5_{\pm2.0}$ | $\mathbf{75.3}_{\pm14.3}/\mathbf{63.2}_{\pm19.7}$ | $51.7_{\pm30.2}$ | $\mathbf{73.7}_{\pm15.6}/\mathbf{70.5}_{\pm17.1}$ | $51.7_{\pm30.2}$ | $79.8_{\pm6.2}/79.1_{\pm7.1}$ | $51.7_{\pm30.2}$ | $\mathbf{49.0}_{\pm5.6}/38.1_{\pm6.1}$ | $83.3_{\pm1.3}$ | $53.3_{\pm4.4}/43.3_{\pm5.9}$ | $94.2_{\pm1.0}$ |
| | GCN-EBM | $88.7_{\pm0.7}/88.2_{\pm0.9}$ | $90.4_{\pm0.9}$ | $80.7_{\pm2.9}/80.0_{\pm3.4}$ | $85.0_{\pm3.2}$ | $50.8_{\pm4.6}/48.5_{\pm4.2}$ | $81.9_{\pm0.8}$ | $54.3_{\pm4.4}/52.3_{\pm4.8}$ | $81.9_{\pm0.8}$ | $77.2_{\pm2.5}/75.6_{\pm3.3}$ | $81.9_{\pm0.8}$ | $45.3_{\pm6.3}/41.7_{\pm6.4}$ | $82.5_{\pm1.6}$ | $52.8_{\pm3.6}/50.4_{\pm3.8}$ | $94.1_{\pm0.6}$ |
| | GCN-HEAT | $88.7_{\pm0.7}/85.2_{\pm1.8}$ | $90.4_{\pm0.9}$ | $80.7_{\pm2.9}/74.6_{\pm6.3}$ | $85.0_{\pm3.2}$ | $50.8_{\pm4.6}/46.1_{\pm3.7}$ | $81.9_{\pm0.8}$ | $54.3_{\pm4.4}/45.3_{\pm4.6}$ | $81.9_{\pm0.8}$ | $77.2_{\pm2.5}/\mathbf{83.3}_{\pm2.4}$ | $81.9_{\pm0.8}$ | $45.3_{\pm6.3}/39.7_{\pm7.9}$ | $82.5_{\pm1.6}$ | $52.8_{\pm3.6}/47.6_{\pm5.2}$ | $94.1_{\pm0.6}$ |
| | GCNSafe | $88.7_{\pm0.7}/\mathbf{90.2}_{\pm1.0}$ | $90.4_{\pm0.9}$ | $80.7_{\pm2.9}/\mathbf{82.8}_{\pm2.5}$ | $85.0_{\pm3.2}$ | $50.8_{\pm4.6}/51.5_{\pm4.1}$ | $81.9_{\pm0.8}$ | $54.3_{\pm4.4}/53.5_{\pm3.9}$ | $81.9_{\pm0.8}$ | $77.2_{\pm2.5}/75.6_{\pm2.9}$ | $81.9_{\pm0.8}$ | $45.3_{\pm6.3}/40.1_{\pm7.1}$ | $82.5_{\pm1.6}$ | $52.8_{\pm3.6}/47.6_{\pm2.5}$ | $94.1_{\pm0.6}$ |
| | **GCN-GEBM** | $88.7_{\pm0.7}/\mathbf{89.8}_{\pm1.2}$ | $90.4_{\pm0.9}$ | $80.7_{\pm2.9}/82.3_{\pm3.3}$ | $85.0_{\pm3.2}$ | $50.8_{\pm4.6}/46.8_{\pm5.3}$ | $81.9_{\pm0.8}$ | $54.3_{\pm4.4}/45.7_{\pm5.5}$ | $81.9_{\pm0.8}$ | $77.2_{\pm2.5}/81.4_{\pm3.0}$ | $81.9_{\pm0.8}$ | $45.3_{\pm6.3}/39.2_{\pm6.6}$ | $82.5_{\pm1.6}$ | $52.8_{\pm3.6}/43.9_{\pm3.6}$ | $94.1_{\pm0.6}$ |
| CoraML-LLM | GCN-MCD | $89.3_{\pm2.7}/50.4_{\pm2.1}$ | $91.2_{\pm0.7}$ | $81.4_{\pm4.0}/39.4_{\pm2.6}$ | $90.6_{\pm2.0}$ | n.a./n.a. | n.a. | $48.7_{\pm3.2}/24.5_{\pm3.8}$ | $82.6_{\pm1.5}$ | $65.2_{\pm2.8}/63.4_{\pm2.9}$ | $82.6_{\pm1.5}$ | $50.1_{\pm5.0}/19.9_{\pm2.0}$ | $84.9_{\pm2.1}$ | $59.7_{\pm4.9}/19.9_{\pm2.2}$ | $94.1_{\pm1.1}$ |
| | GCN-DE | $88.7_{\pm1.7}/73.9_{\pm2.5}$ | $89.7_{\pm1.2}$ | $80.6_{\pm4.6}/62.2_{\pm3.7}$ | $88.9_{\pm1.7}$ | n.a./n.a. | n.a. | $53.1_{\pm4.5}/33.8_{\pm3.9}$ | $79.9_{\pm1.8}$ | $66.7_{\pm3.2}/\mathbf{77.4}_{\pm2.6}$ | $79.9_{\pm1.8}$ | $51.8_{\pm5.7}/37.1_{\pm3.4}$ | $82.2_{\pm2.7}$ | $55.7_{\pm5.1}/39.8_{\pm4.3}$ | $91.7_{\pm1.4}$ |
| | GCN-MCD+DE | $89.2_{\pm1.8}/73.9_{\pm2.6}$ | $89.4_{\pm0.9}$ | $80.8_{\pm3.6}/61.4_{\pm4.6}$ | $88.9_{\pm1.7}$ | n.a./n.a. | n.a. | $\mathbf{55.0}_{\pm4.0}/33.6_{\pm3.5}$ | $80.3_{\pm1.8}$ | $66.0_{\pm3.0}/76.2_{\pm2.6}$ | $80.3_{\pm1.8}$ | $51.5_{\pm5.1}/38.2_{\pm4.1}$ | $82.7_{\pm2.6}$ | $58.3_{\pm3.9}/40.1_{\pm3.6}$ | $91.4_{\pm1.5}$ |
| | GCN-BNN | $89.3_{\pm1.8}/82.7_{\pm3.4}$ | $91.4_{\pm0.8}$ | $80.1_{\pm5.1}/73.6_{\pm5.6}$ | $91.0_{\pm2.0}$ | n.a./n.a. | n.a. | $47.8_{\pm5.0}/28.9_{\pm3.6}$ | $82.8_{\pm1.6}$ | $68.1_{\pm2.4}/73.7_{\pm3.0}$ | $82.8_{\pm1.6}$ | $51.6_{\pm7.8}/42.3_{\pm5.6}$ | $84.0_{\pm2.2}$ | $\mathbf{61.1}_{\pm5.7}/53.7_{\pm6.2}$ | $94.0_{\pm1.3}$ |
| | GCN-Ens | $89.4_{\pm2.5}/83.8_{\pm1.8}$ | $91.4_{\pm0.7}$ | $81.6_{\pm4.2}/67.0_{\pm6.3}$ | $90.6_{\pm2.3}$ | n.a./n.a. | n.a. | $49.1_{\pm3.4}/27.5_{\pm6.0}$ | $82.6_{\pm1.5}$ | $66.0_{\pm4.2}/\mathbf{76.4}_{\pm3.0}$ | $82.6_{\pm1.5}$ | $49.5_{\pm4.6}/32.8_{\pm3.2}$ | $85.0_{\pm2.1}$ | $60.3_{\pm5.2}/44.0_{\pm5.3}$ | $94.0_{\pm1.1}$ |
| | GPN | $87.4_{\pm2.3}/89.3_{\pm1.4}$ | $90.5_{\pm1.1}$ | $82.2_{\pm6.2}/76.5_{\pm4.8}$ | $88.5_{\pm2.2}$ | n.a./n.a. | n.a. | $53.7_{\pm5.3}/42.6_{\pm7.3}$ | $79.7_{\pm1.8}$ | $53.2_{\pm5.1}/42.0_{\pm4.7}$ | $79.7_{\pm1.8}$ | $\mathbf{54.6}_{\pm5.9}/39.1_{\pm5.4}$ | $80.9_{\pm2.2}$ | $58.1_{\pm4.8}/44.3_{\pm6.4}$ | $92.5_{\pm1.0}$ |
| | SGCN | $\mathbf{90.9}_{\pm1.7}/90.4_{\pm1.2}$ | $90.3_{\pm0.9}$ | $83.1_{\pm3.8}/\mathbf{83.3}_{\pm4.7}$ | $90.6_{\pm1.6}$ | n.a./n.a. | n.a. | $\mathbf{65.1}_{\pm11.4}/43.5_{\pm16.1}$ | $68.7_{\pm21.0}$ | $69.4_{\pm8.1}/67.0_{\pm10.1}$ | $68.7_{\pm21.0}$ | $\mathbf{54.4}_{\pm6.3}/39.3_{\pm8.2}$ | $82.0_{\pm2.7}$ | $56.7_{\pm4.1}/46.4_{\pm4.9}$ | $93.1_{\pm1.1}$ |
| | GCN-EBM | $89.2_{\pm2.4}/88.3_{\pm3.0}$ | $91.4_{\pm0.7}$ | $81.5_{\pm4.1}/81.4_{\pm4.1}$ | $90.7_{\pm2.0}$ | n.a./n.a. | n.a. | $47.6_{\pm3.1}/43.7_{\pm4.0}$ | $82.6_{\pm1.6}$ | $67.1_{\pm2.3}/64.6_{\pm2.2}$ | $82.6_{\pm1.6}$ | $50.4_{\pm5.1}/44.9_{\pm3.6}$ | $85.0_{\pm2.2}$ | $\mathbf{60.9}_{\pm4.8}/57.4_{\pm5.2}$ | $93.8_{\pm1.0}$ |
| | GCN-HEAT | $89.2_{\pm2.4}/82.5_{\pm6.2}$ | $91.4_{\pm0.7}$ | $81.5_{\pm4.1}/74.2_{\pm8.2}$ | $90.7_{\pm2.0}$ | n.a./n.a. | n.a. | $47.6_{\pm3.1}/38.4_{\pm6.1}$ | $82.6_{\pm1.6}$ | $67.1_{\pm2.3}/72.8_{\pm2.9}$ | $82.6_{\pm1.6}$ | $50.4_{\pm5.1}/36.6_{\pm6.5}$ | $85.0_{\pm2.2}$ | $\mathbf{60.9}_{\pm4.8}/46.3_{\pm4.9}$ | $93.8_{\pm1.0}$ |
| | GCNSafe | $89.2_{\pm2.4}/\mathbf{90.6}_{\pm2.7}$ | $91.4_{\pm0.7}$ | $81.5_{\pm4.1}/\mathbf{86.4}_{\pm2.9}$ | $90.7_{\pm2.0}$ | n.a./n.a. | n.a. | $47.6_{\pm3.1}/50.2_{\pm3.6}$ | $82.6_{\pm1.6}$ | $67.1_{\pm2.3}/65.2_{\pm2.5}$ | $82.6_{\pm1.6}$ | $50.4_{\pm5.1}/45.3_{\pm3.8}$ | $85.0_{\pm2.2}$ | $\mathbf{60.9}_{\pm4.8}/52.2_{\pm4.8}$ | $93.8_{\pm1.0}$ |
| | **GCN-GEBM** | $89.2_{\pm2.4}/89.6_{\pm2.1}$ | $91.4_{\pm0.7}$ | $81.5_{\pm4.1}/81.6_{\pm6.9}$ | $90.7_{\pm2.0}$ | n.a./n.a. | n.a. | $47.6_{\pm3.1}/37.4_{\pm4.3}$ | $82.6_{\pm1.6}$ | $67.1_{\pm2.3}/70.6_{\pm3.6}$ | $82.6_{\pm1.6}$ | $50.4_{\pm5.1}/40.5_{\pm6.0}$ | $85.0_{\pm2.2}$ | $\mathbf{60.9}_{\pm4.8}/50.9_{\pm6.0}$ | $93.8_{\pm1.0}$ |
| Citeseer | GCN-MCD | $81.8_{\pm1.0}/41.9_{\pm2.2}$ | $83.5_{\pm1.2}$ | $82.8_{\pm2.6}/42.7_{\pm2.0}$ | $91.1_{\pm0.8}$ | $56.2_{\pm4.3}/36.7_{\pm3.1}$ | $82.1_{\pm1.0}$ | $61.1_{\pm3.9}/39.0_{\pm5.1}$ | $82.1_{\pm1.0}$ | $69.0_{\pm3.3}/71.0_{\pm2.4}$ | $82.1_{\pm1.0}$ | $47.0_{\pm6.7}/19.5_{\pm3.2}$ | $82.9_{\pm1.1}$ | $50.2_{\pm7.2}/19.2_{\pm2.6}$ | $86.3_{\pm1.5}$ |
| | GCN-DE | $79.0_{\pm2.2}/66.7_{\pm3.3}$ | $80.1_{\pm1.7}$ | $81.2_{\pm2.4}/66.7_{\pm2.7}$ | $86.4_{\pm1.0}$ | $56.3_{\pm2.4}/\mathbf{62.6}_{\pm2.2}$ | $76.8_{\pm1.5}$ | $57.8_{\pm3.1}/\mathbf{73.8}_{\pm2.6}$ | $76.8_{\pm1.5}$ | $65.3_{\pm2.7}/77.9_{\pm2.0}$ | $76.8_{\pm1.5}$ | $47.4_{\pm4.9}/39.5_{\pm3.9}$ | $79.0_{\pm1.3}$ | $49.3_{\pm4.4}/45.5_{\pm3.3}$ | $81.6_{\pm1.6}$ |
| | GCN-MCD+DE | $78.8_{\pm2.1}/66.1_{\pm2.6}$ | $80.2_{\pm1.7}$ | $81.2_{\pm2.4}/68.3_{\pm2.7}$ | $86.2_{\pm1.1}$ | $55.8_{\pm2.4}/61.7_{\pm3.4}$ | $76.8_{\pm1.8}$ | $58.5_{\pm4.1}/\mathbf{74.3}_{\pm3.1}$ | $76.8_{\pm1.8}$ | $65.6_{\pm2.4}/78.8_{\pm1.8}$ | $76.8_{\pm1.8}$ | $48.1_{\pm4.9}/42.5_{\pm4.4}$ | $79.2_{\pm1.0}$ | $50.7_{\pm3.9}/45.9_{\pm2.9}$ | $81.7_{\pm1.5}$ |
| | GCN-BNN | $80.2_{\pm2.0}/79.3_{\pm2.6}$ | $82.5_{\pm1.4}$ | $81.4_{\pm2.5}/80.4_{\pm1.8}$ | $90.2_{\pm0.7}$ | $55.3_{\pm2.3}/62.2_{\pm2.0}$ | $81.2_{\pm1.3}$ | $63.4_{\pm4.2}/73.5_{\pm5.6}$ | $81.2_{\pm1.3}$ | $71.9_{\pm2.7}/\mathbf{79.7}_{\pm1.7}$ | $81.2_{\pm1.3}$ | $46.8_{\pm6.7}/47.8_{\pm5.8}$ | $81.3_{\pm1.1}$ | $50.0_{\pm3.2}/50.3_{\pm4.2}$ | $86.6_{\pm1.1}$ |
| | GCN-Ens | $81.6_{\pm0.6}/70.0_{\pm2.5}$ | $83.8_{\pm1.1}$ | $82.6_{\pm2.5}/73.4_{\pm3.2}$ | $91.2_{\pm0.8}$ | $54.0_{\pm1.5}/49.3_{\pm1.3}$ | $82.0_{\pm1.0}$ | $55.7_{\pm4.2}/60.9_{\pm5.7}$ | $82.0_{\pm1.0}$ | $69.2_{\pm2.7}/77.5_{\pm1.8}$ | $82.0_{\pm1.0}$ | $47.1_{\pm5.3}/35.8_{\pm3.0}$ | $82.6_{\pm0.8}$ | $49.2_{\pm7.5}/49.4_{\pm4.6}$ | $86.4_{\pm1.6}$ |
| | GPN | $81.5_{\pm4.1}/80.5_{\pm3.7}$ | $84.9_{\pm1.9}$ | $\mathbf{86.0}_{\pm3.1}/78.9_{\pm3.8}$ | $90.9_{\pm0.9}$ | $60.1_{\pm5.5}/57.9_{\pm4.8}$ | $83.3_{\pm1.9}$ | $60.1_{\pm4.6}/69.8_{\pm6.2}$ | $83.3_{\pm1.9}$ | $67.6_{\pm3.7}/62.3_{\pm5.4}$ | $83.3_{\pm1.9}$ | $\mathbf{54.5}_{\pm6.1}/44.7_{\pm8.1}$ | $82.1_{\pm1.5}$ | $\mathbf{55.0}_{\pm4.7}/47.2_{\pm5.0}$ | $88.0_{\pm1.0}$ |
| | SGCN | $82.8_{\pm2.9}/\mathbf{85.5}_{\pm2.8}$ | $83.5_{\pm1.9}$ | $84.8_{\pm2.3}/\mathbf{88.6}_{\pm2.7}$ | $90.6_{\pm0.6}$ | $60.4_{\pm2.7}/54.8_{\pm3.7}$ | $81.5_{\pm1.8}$ | $65.9_{\pm3.4}/61.4_{\pm3.9}$ | $81.5_{\pm1.8}$ | $69.2_{\pm2.7}/68.3_{\pm3.0}$ | $81.5_{\pm1.8}$ | $48.5_{\pm4.2}/43.3_{\pm4.3}$ | $82.5_{\pm1.0}$ | $51.0_{\pm6.1}/48.7_{\pm5.6}$ | $85.9_{\pm1.4}$ |
| | GCN-EBM | $81.6_{\pm1.1}/81.2_{\pm1.5}$ | $83.6_{\pm1.1}$ | $82.3_{\pm2.7}/81.3_{\pm3.2}$ | $91.1_{\pm0.9}$ | $55.6_{\pm3.6}/54.4_{\pm4.0}$ | $82.0_{\pm1.0}$ | $60.0_{\pm3.2}/53.6_{\pm4.2}$ | $82.0_{\pm1.0}$ | $70.1_{\pm1.3}/66.9_{\pm1.6}$ | $82.0_{\pm1.0}$ | $47.3_{\pm6.2}/45.1_{\pm6.0}$ | $82.4_{\pm1.0}$ | $49.2_{\pm6.6}/46.6_{\pm5.9}$ | $86.4_{\pm1.5}$ |
| | GCN-HEAT | $81.6_{\pm1.1}/74.6_{\pm2.5}$ | $83.6_{\pm1.1}$ | $82.3_{\pm2.7}/75.2_{\pm3.1}$ | $91.1_{\pm0.9}$ | $55.6_{\pm3.6}/47.5_{\pm2.6}$ | $82.0_{\pm1.0}$ | $60.0_{\pm3.2}/56.0_{\pm4.8}$ | $82.0_{\pm1.0}$ | $70.1_{\pm1.3}/\mathbf{79.2}_{\pm2.3}$ | $82.0_{\pm1.0}$ | $47.3_{\pm6.2}/38.5_{\pm3.9}$ | $82.4_{\pm1.0}$ | $49.2_{\pm6.6}/38.8_{\pm5.0}$ | $86.4_{\pm1.5}$ |
| | GCNSafe | $81.6_{\pm1.1}/\mathbf{84.2}_{\pm1.8}$ | $83.6_{\pm1.1}$ | $82.3_{\pm2.7}/81.9_{\pm3.9}$ | $91.1_{\pm0.9}$ | $55.6_{\pm3.6}/\mathbf{63.1}_{\pm3.9}$ | $82.0_{\pm1.0}$ | $60.0_{\pm3.2}/62.8_{\pm2.4}$ | $82.0_{\pm1.0}$ | $70.1_{\pm1.3}/66.2_{\pm1.4}$ | $82.0_{\pm1.0}$ | $47.3_{\pm6.2}/\mathbf{49.9}_{\pm5.9}$ | $82.4_{\pm1.0}$ | $49.2_{\pm6.6}/\mathbf{52.5}_{\pm4.7}$ | $86.4_{\pm1.5}$ |
| | **GCN-GEBM** | $81.6_{\pm1.1}/82.8_{\pm2.8}$ | $83.6_{\pm1.1}$ | $82.3_{\pm2.7}/82.3_{\pm3.2}$ | $91.1_{\pm0.9}$ | $55.6_{\pm3.6}/57.2_{\pm5.7}$ | $82.0_{\pm1.0}$ | $60.0_{\pm3.2}/62.7_{\pm5.5}$ | $82.0_{\pm1.0}$ | $70.1_{\pm1.3}/78.0_{\pm1.7}$ | $82.0_{\pm1.0}$ | $47.3_{\pm6.2}/46.0_{\pm5.5}$ | $82.4_{\pm1.0}$ | $49.2_{\pm6.6}/49.9_{\pm6.5}$ | $86.4_{\pm1.5}$ |
| PubMed | GCN-MCD | $66.3_{\pm1.7}/43.8_{\pm1.2}$ | $91.6_{\pm1.6}$ | $53.4_{\pm3.4}/32.3_{\pm1.1}$ | $83.7_{\pm1.5}$ | n.a./n.a. | n.a. | $47.6_{\pm2.8}/30.5_{\pm3.1}$ | $78.1_{\pm1.0}$ | $57.4_{\pm1.2}/56.0_{\pm1.5}$ | $78.1_{\pm1.0}$ | $41.3_{\pm2.4}/21.8_{\pm2.2}$ | $77.9_{\pm2.6}$ | $42.3_{\pm3.0}/20.2_{\pm2.6}$ | $88.9_{\pm2.8}$ |
| | GCN-DE | $65.9_{\pm1.8}/55.2_{\pm2.3}$ | $91.2_{\pm1.7}$ | $54.8_{\pm2.2}/47.5_{\pm1.8}$ | $80.5_{\pm2.1}$ | n.a./n.a. | n.a. | $47.6_{\pm2.8}/\mathbf{51.6}_{\pm1.7}$ | $74.9_{\pm2.1}$ | $55.5_{\pm0.9}/60.2_{\pm1.1}$ | $74.9_{\pm2.1}$ | $40.5_{\pm2.5}/33.5_{\pm2.2}$ | $76.0_{\pm2.6}$ | $\mathbf{42.6}_{\pm2.9}/36.2_{\pm2.0}$ | $84.8_{\pm2.3}$ |
| | GCN-MCD+DE | $\mathbf{66.6}_{\pm2.3}/55.4_{\pm2.6}$ | $90.8_{\pm2.1}$ | $55.5_{\pm2.5}/47.9_{\pm1.6}$ | $80.7_{\pm1.8}$ | n.a./n.a. | n.a. | $47.0_{\pm2.2}/\mathbf{51.6}_{\pm1.8}$ | $75.4_{\pm1.5}$ | $55.0_{\pm1.3}/59.4_{\pm1.3}$ | $75.4_{\pm1.5}$ | $40.1_{\pm2.8}/33.7_{\pm2.0}$ | $76.3_{\pm1.9}$ | $42.5_{\pm2.6}/36.0_{\pm1.7}$ | $84.9_{\pm1.8}$ |
| | GCN-BNN | $66.5_{\pm3.7}/64.6_{\pm4.1}$ | $90.7_{\pm1.6}$ | $53.4_{\pm3.7}/52.7_{\pm4.3}$ | $84.0_{\pm1.7}$ | n.a./n.a. | n.a. | $\mathbf{51.2}_{\pm3.0}/47.5_{\pm4.8}$ | $78.2_{\pm1.6}$ | $59.1_{\pm2.4}/\mathbf{61.5}_{\pm2.2}$ | $78.2_{\pm1.6}$ | $\mathbf{43.5}_{\pm3.3}/40.9_{\pm3.0}$ | $76.1_{\pm4.2}$ | $\mathbf{42.6}_{\pm2.7}/41.8_{\pm2.8}$ | $89.3_{\pm1.9}$ |
| | GCN-Ens | $66.2_{\pm1.8}/50.9_{\pm3.4}$ | $91.7_{\pm1.5}$ | $53.3_{\pm3.5}/34.0_{\pm3.6}$ | $83.8_{\pm1.4}$ | n.a./n.a. | n.a. | $47.5_{\pm2.7}/36.7_{\pm6.3}$ | $78.1_{\pm0.9}$ | $58.4_{\pm0.6}/61.4_{\pm1.0}$ | $78.1_{\pm0.9}$ | $41.4_{\pm2.7}/23.9_{\pm2.6}$ | $77.9_{\pm2.6}$ | $42.1_{\pm3.0}/30.2_{\pm3.5}$ | $88.9_{\pm2.8}$ |
| | GPN | $64.9_{\pm2.3}/62.6_{\pm6.1}$ | $92.3_{\pm0.8}$ | $53.1_{\pm4.0}/55.1_{\pm5.2}$ | $85.6_{\pm1.2}$ | n.a./n.a. | n.a. | $48.3_{\pm4.0}/38.6_{\pm7.0}$ | $79.8_{\pm1.1}$ | $43.6_{\pm1.7}/30.6_{\pm1.9}$ | $79.8_{\pm1.1}$ | $\mathbf{43.2}_{\pm2.5}/31.5_{\pm2.6}$ | $80.2_{\pm1.4}$ | $\mathbf{43.6}_{\pm2.1}/32.9_{\pm2.1}$ | $90.3_{\pm1.7}$ |
| | SGCN | $66.3_{\pm1.7}/\mathbf{68.6}_{\pm2.3}$ | $92.1_{\pm1.6}$ | $54.3_{\pm3.5}/\mathbf{56.4}_{\pm4.0}$ | $85.1_{\pm0.9}$ | n.a./n.a. | n.a. | $48.1_{\pm2.9}/39.0_{\pm2.4}$ | $80.1_{\pm0.9}$ | $57.2_{\pm1.1}/56.1_{\pm1.6}$ | $80.1_{\pm0.9}$ | $42.0_{\pm3.0}/34.4_{\pm4.8}$ | $79.3_{\pm2.8}$ | $41.3_{\pm2.5}/36.0_{\pm3.4}$ | $90.7_{\pm1.8}$ |
| | GCN-EBM | $66.2_{\pm1.7}/66.3_{\pm1.9}$ | $91.7_{\pm1.5}$ | $53.4_{\pm3.6}/52.9_{\pm3.7}$ | $83.7_{\pm1.5}$ | n.a./n.a. | n.a. | $48.6_{\pm2.9}/43.2_{\pm2.9}$ | $78.1_{\pm1.0}$ | $58.5_{\pm1.4}/58.3_{\pm1.5}$ | $78.1_{\pm1.0}$ | $41.2_{\pm2.7}/38.4_{\pm3.4}$ | $78.0_{\pm2.6}$ | $\mathbf{42.6}_{\pm3.1}/40.2_{\pm3.2}$ | $88.8_{\pm2.7}$ |
| | GCN-HEAT | $66.2_{\pm1.7}/59.9_{\pm3.4}$ | $91.7_{\pm1.5}$ | $53.4_{\pm3.6}/45.6_{\pm4.8}$ | $83.7_{\pm1.5}$ | n.a./n.a. | n.a. | $48.6_{\pm2.9}/36.0_{\pm3.7}$ | $78.1_{\pm1.0}$ | $58.5_{\pm1.4}/\mathbf{62.0}_{\pm2.0}$ | $78.1_{\pm1.0}$ | $41.2_{\pm2.7}/30.9_{\pm5.4}$ | $78.0_{\pm2.6}$ | $\mathbf{42.6}_{\pm3.1}/33.0_{\pm4.2}$ | $88.8_{\pm2.7}$ |
| | GCNSafe | $66.2_{\pm1.7}/65.3_{\pm1.9}$ | $91.7_{\pm1.5}$ | $53.4_{\pm3.6}/\mathbf{56.7}_{\pm4.4}$ | $83.7_{\pm1.5}$ | n.a./n.a. | n.a. | $48.6_{\pm2.9}/46.1_{\pm3.6}$ | $78.1_{\pm1.0}$ | $58.5_{\pm1.4}/57.8_{\pm1.3}$ | $78.1_{\pm1.0}$ | $41.2_{\pm2.7}/37.2_{\pm3.3}$ | $78.0_{\pm2.6}$ | $\mathbf{42.6}_{\pm3.1}/37.7_{\pm4.2}$ | $88.8_{\pm2.7}$ |
| | **GCN-GEBM** | $66.2_{\pm1.7}/63.2_{\pm3.2}$ | $91.7_{\pm1.5}$ | $53.4_{\pm3.6}/51.6_{\pm4.0}$ | $83.7_{\pm1.5}$ | n.a./n.a. | n.a. | $48.6_{\pm2.9}/39.4_{\pm5.1}$ | $78.1_{\pm1.0}$ | $58.5_{\pm1.4}/61.1_{\pm1.9}$ | $78.1_{\pm1.0}$ | $41.2_{\pm2.7}/28.4_{\pm3.0}$ | $78.0_{\pm2.6}$ | $\mathbf{42.6}_{\pm3.1}/32.8_{\pm5.6}$ | $88.8_{\pm2.7}$ |
| Amazon Computers | GCN-MCD | $68.4_{\pm6.8}/37.8_{\pm1.2}$ | $88.0_{\pm1.5}$ | $84.7_{\pm3.0}/45.4_{\pm1.9}$ | $93.5_{\pm2.4}$ | $35.0_{\pm2.5}/22.3_{\pm2.6}$ | $81.7_{\pm1.1}$ | $49.4_{\pm10.9}/43.7_{\pm11.8}$ | $81.7_{\pm1.1}$ | $69.0_{\pm4.5}/68.0_{\pm5.1}$ | $81.7_{\pm1.1}$ | $34.5_{\pm2.7}/19.5_{\pm1.3}$ | $82.7_{\pm1.1}$ | $36.3_{\pm1.9}/18.9_{\pm1.3}$ | $92.3_{\pm0.7}$ |
| | GCN-DE | $67.3_{\pm6.3}/66.2_{\pm3.1}$ | $87.2_{\pm1.2}$ | $\mathbf{84.9}_{\pm2.5}/73.9_{\pm3.1}$ | $93.0_{\pm2.7}$ | $34.9_{\pm2.9}/45.0_{\pm1.9}$ | $80.7_{\pm1.2}$ | $47.8_{\pm12.4}/55.4_{\pm4.1}$ | $80.7_{\pm1.2}$ | $69.9_{\pm3.7}/\mathbf{81.8}_{\pm1.4}$ | $80.7_{\pm1.2}$ | $34.2_{\pm3.0}/41.2_{\pm2.9}$ | $82.0_{\pm1.0}$ | $36.0_{\pm2.6}/38.9_{\pm2.0}$ | $90.9_{\pm0.8}$ |
| | GCN-MCD+DE | $67.9_{\pm5.6}/65.7_{\pm3.0}$ | $87.1_{\pm1.2}$ | $\mathbf{85.2}_{\pm3.0}/74.8_{\pm3.3}$ | $92.8_{\pm2.7}$ | $35.4_{\pm2.5}/45.5_{\pm1.6}$ | $80.7_{\pm1.3}$ | $49.2_{\pm10.4}/55.2_{\pm4.0}$ | $80.7_{\pm1.3}$ | $69.8_{\pm4.4}/81.4_{\pm2.1}$ | $80.7_{\pm1.3}$ | $34.3_{\pm2.9}/41.4_{\pm2.9}$ | $81.7_{\pm1.5}$ | $36.7_{\pm2.2}/39.0_{\pm2.0}$ | $91.1_{\pm0.7}$ |
| | GCN-BNN | $64.1_{\pm8.2}/69.0_{\pm8.5}$ | $85.4_{\pm3.2}$ | $82.5_{\pm5.7}/84.1_{\pm5.9}$ | $92.3_{\pm3.4}$ | $46.5_{\pm8.8}/45.6_{\pm8.8}$ | $77.6_{\pm2.6}$ | $65.8_{\pm16.0}/63.9_{\pm12.3}$ | $77.6_{\pm2.6}$ | $68.1_{\pm2.5}/79.1_{\pm1.6}$ | $77.6_{\pm2.6}$ | $36.3_{\pm3.1}/44.6_{\pm2.9}$ | $80.7_{\pm2.4}$ | $40.6_{\pm4.0}/45.6_{\pm4.1}$ | $90.9_{\pm1.6}$ |
| | GCN-Ens | $68.5_{\pm7.1}/61.4_{\pm6.2}$ | $88.1_{\pm1.7}$ | $84.5_{\pm3.2}/80.7_{\pm5.8}$ | $93.5_{\pm2.6}$ | $34.6_{\pm3.4}/32.7_{\pm3.0}$ | $81.7_{\pm0.9}$ | $49.8_{\pm11.5}/48.0_{\pm9.9}$ | $81.7_{\pm0.9}$ | $69.9_{\pm4.4}/79.2_{\pm1.5}$ | $81.7_{\pm0.9}$ | $34.4_{\pm2.4}/34.4_{\pm2.5}$ | $82.6_{\pm1.0}$ | $35.4_{\pm1.4}/42.1_{\pm2.2}$ | $92.4_{\pm0.6}$ |
| | GPN | $\mathbf{80.6}_{\pm3.2}/70.7_{\pm4.3}$ | $86.0_{\pm1.4}$ | $81.7_{\pm3.4}/77.8_{\pm3.3}$ | $94.4_{\pm1.1}$ | $48.3_{\pm6.1}/38.1_{\pm3.3}$ | $79.5_{\pm2.4}$ | $48.3_{\pm5.2}/39.2_{\pm3.4}$ | $79.5_{\pm2.4}$ | $52.8_{\pm4.3}/40.3_{\pm1.9}$ | $79.5_{\pm2.4}$ | $47.7_{\pm4.3}/40.0_{\pm2.6}$ | $79.3_{\pm2.0}$ | $50.0_{\pm7.6}/40.3_{\pm8.9}$ | $86.1_{\pm14.4}$ |
| | SGCN | $70.1_{\pm11.8}/75.7_{\pm12.0}$ | $58.0_{\pm29.3}$ | $74.9_{\pm8.2}/80.4_{\pm6.1}$ | $92.7_{\pm6.5}$ | $\mathbf{82.4}_{\pm17.6}/\mathbf{85.3}_{\pm13.1}$ | $16.1_{\pm15.1}$ | $\mathbf{82.6}_{\pm17.6}/\mathbf{85.3}_{\pm13.2}$ | $16.1_{\pm15.1}$ | $81.5_{\pm15.1}/\mathbf{85.6}_{\pm11.1}$ | $16.1_{\pm15.1}$ | $\mathbf{81.1}_{\pm15.3}/\mathbf{86.2}_{\pm10.5}$ | $14.9_{\pm13.4}$ | $\mathbf{78.1}_{\pm13.9}/\mathbf{89.8}_{\pm8.4}$ | $16.7_{\pm10.6}$ |
| | GCN-EBM | $68.5_{\pm6.8}/65.2_{\pm5.2}$ | $88.0_{\pm1.8}$ | $84.4_{\pm3.2}/81.3_{\pm3.9}$ | $93.5_{\pm2.5}$ | $34.7_{\pm2.7}/33.3_{\pm3.5}$ | $81.9_{\pm1.0}$ | $49.1_{\pm12.0}/48.5_{\pm12.5}$ | $81.9_{\pm1.0}$ | $65.9_{\pm3.9}/65.8_{\pm4.9}$ | $81.9_{\pm1.0}$ | $34.3_{\pm3.3}/32.3_{\pm3.0}$ | $82.6_{\pm1.2}$ | $35.5_{\pm1.6}/31.6_{\pm1.7}$ | $92.4_{\pm0.7}$ |
| | GCN-HEAT | $68.5_{\pm6.8}/59.0_{\pm6.2}$ | $88.0_{\pm1.8}$ | $84.4_{\pm3.2}/75.3_{\pm4.2}$ | $93.5_{\pm2.5}$ | $34.7_{\pm2.7}/30.3_{\pm5.1}$ | $81.9_{\pm1.0}$ | $49.1_{\pm12.0}/50.1_{\pm12.8}$ | $81.9_{\pm1.0}$ | $65.9_{\pm3.9}/76.9_{\pm2.6}$ | $81.9_{\pm1.0}$ | $34.3_{\pm3.3}/27.6_{\pm3.5}$ | $82.6_{\pm1.2}$ | $35.5_{\pm1.6}/28.4_{\pm2.3}$ | $92.4_{\pm0.7}$ |
| | GCNSafe | $68.5_{\pm6.8}/\mathbf{79.2}_{\pm3.2}$ | $88.0_{\pm1.8}$ | $84.4_{\pm3.2}/81.8_{\pm3.1}$ | $93.5_{\pm2.5}$ | $34.7_{\pm2.7}/40.6_{\pm3.9}$ | $81.9_{\pm1.0}$ | $49.1_{\pm12.0}/52.5_{\pm9.1}$ | $81.9_{\pm1.0}$ | $65.9_{\pm3.9}/69.4_{\pm4.1}$ | $81.9_{\pm1.0}$ | $34.3_{\pm3.3}/42.5_{\pm3.4}$ | $82.6_{\pm1.2}$ | $35.5_{\pm1.6}/39.2_{\pm1.7}$ | $92.4_{\pm0.7}$ |
| | **GCN-GEBM** | $68.5_{\pm6.8}/77.8_{\pm2.6}$ | $88.0_{\pm1.8}$ | $84.4_{\pm3.2}/84.1_{\pm2.1}$ | $93.5_{\pm2.5}$ | $34.7_{\pm2.7}/25.6_{\pm4.9}$ | $81.9_{\pm1.0}$ | $49.1_{\pm12.0}/46.6_{\pm17.2}$ | $81.9_{\pm1.0}$ | $65.9_{\pm3.9}/69.0_{\pm3.7}$ | $81.9_{\pm1.0}$ | $34.3_{\pm3.3}/32.0_{\pm4.5}$ | $82.6_{\pm1.2}$ | $35.5_{\pm1.6}/28.0_{\pm3.7}$ | $92.4_{\pm0.7}$ |
| Amazon Photo | GCN-MCD | $74.3_{\pm5.0}/46.3_{\pm1.3}$ | $92.0_{\pm0.9}$ | $66.9_{\pm6.0}/31.0_{\pm1.3}$ | $96.3_{\pm0.2}$ | $38.1_{\pm5.7}/17.6_{\pm4.0}$ | $90.2_{\pm2.0}$ | $64.7_{\pm5.6}/47.6_{\pm8.1}$ | $90.2_{\pm2.0}$ | $65.9_{\pm4.4}/61.1_{\pm6.7}$ | $90.2_{\pm2.0}$ | $41.0_{\pm5.8}/14.4_{\pm2.6}$ | $91.8_{\pm0.6}$ | $46.1_{\pm4.0}/12.6_{\pm2.0}$ | $97.6_{\pm0.7}$ |
| | GCN-DE | $75.7_{\pm4.8}/57.1_{\pm2.1}$ | $91.3_{\pm1.1}$ | $66.2_{\pm4.7}/50.6_{\pm3.1}$ | $95.9_{\pm0.4}$ | $36.4_{\pm3.8}/37.9_{\pm3.5}$ | $89.5_{\pm1.9}$ | $61.4_{\pm6.6}/46.7_{\pm4.9}$ | $89.5_{\pm1.9}$ | $69.5_{\pm4.9}/76.4_{\pm1.7}$ | $89.5_{\pm1.9}$ | $40.5_{\pm5.3}/27.7_{\pm4.7}$ | $91.2_{\pm0.7}$ | $41.1_{\pm4.9}/29.8_{\pm3.1}$ | $96.9_{\pm0.8}$ |
| | GCN-MCD+DE | $74.9_{\pm5.5}/57.2_{\pm2.5}$ | $91.5_{\pm1.0}$ | $66.2_{\pm5.2}/50.3_{\pm3.0}$ | $95.9_{\pm0.5}$ | $37.0_{\pm4.5}/38.9_{\pm2.5}$ | $89.5_{\pm1.7}$ | $62.2_{\pm5.8}/47.0_{\pm4.6}$ | $89.5_{\pm1.7}$ | $69.9_{\pm5.2}/75.6_{\pm2.3}$ | $89.5_{\pm1.7}$ | $39.4_{\pm4.2}/27.9_{\pm4.3}$ | $91.2_{\pm0.7}$ | $42.0_{\pm4.2}/29.8_{\pm3.0}$ | $96.9_{\pm0.8}$ |
| | GCN-BNN | $77.5_{\pm6.3}/78.2_{\pm8.6}$ | $93.2_{\pm0.8}$ | $73.2_{\pm7.5}/\mathbf{79.2}_{\pm7.7}$ | $96.5_{\pm0.4}$ | $49.8_{\pm5.9}/37.3_{\pm10.4}$ | $90.0_{\pm2.6}$ | $65.4_{\pm14.2}/59.6_{\pm13.9}$ | $90.0_{\pm2.6}$ | $64.3_{\pm2.8}/73.2_{\pm2.6}$ | $90.0_{\pm2.6}$ | $40.4_{\pm3.5}/30.4_{\pm3.8}$ | $90.7_{\pm1.4}$ | $47.1_{\pm4.0}/40.2_{\pm4.9}$ | $97.9_{\pm0.9}$ |
| | GCN-Ens | $74.7_{\pm5.4}/71.7_{\pm8.3}$ | $92.0_{\pm0.8}$ | $66.7_{\pm5.7}/68.1_{\pm8.2}$ | $96.3_{\pm0.2}$ | $39.3_{\pm3.8}/19.3_{\pm3.0}$ | $90.4_{\pm2.0}$ | $64.8_{\pm5.2}/52.0_{\pm14.7}$ | $90.4_{\pm2.0}$ | $66.6_{\pm4.5}/72.5_{\pm3.4}$ | $90.4_{\pm2.0}$ | $40.9_{\pm6.2}/23.8_{\pm3.8}$ | $91.9_{\pm0.5}$ | $46.0_{\pm4.1}/34.6_{\pm3.1}$ | $97.7_{\pm0.6}$ |
| | GPN | $79.4_{\pm6.2}/\mathbf{86.0}_{\pm2.9}$ | $89.5_{\pm1.2}$ | $\mathbf{82.1}_{\pm2.9}/57.1_{\pm7.0}$ | $94.3_{\pm1.6}$ | $46.5_{\pm7.4}/28.7_{\pm5.8}$ | $86.2_{\pm2.6}$ | $46.1_{\pm7.9}/29.2_{\pm5.0}$ | $86.2_{\pm2.6}$ | $51.4_{\pm6.4}/29.9_{\pm5.2}$ | $86.2_{\pm2.6}$ | $44.0_{\pm6.2}/25.4_{\pm4.3}$ | $88.8_{\pm1.9}$ | $42.1_{\pm4.4}/27.2_{\pm4.2}$ | $96.4_{\pm0.9}$ |
| | SGCN | $74.8_{\pm4.7}/78.5_{\pm5.9}$ | $91.9_{\pm0.9}$ | $60.1_{\pm7.8}/59.0_{\pm8.7}$ | $96.0_{\pm1.0}$ | $\mathbf{90.1}_{\pm7.0}/\mathbf{88.2}_{\pm7.3}$ | $11.4_{\pm7.5}$ | $\mathbf{90.8}_{\pm5.5}/\mathbf{88.7}_{\pm6.5}$ | $11.4_{\pm7.5}$ | $\mathbf{89.1}_{\pm6.8}/\mathbf{88.2}_{\pm6.8}$ | $11.4_{\pm7.5}$ | $\mathbf{59.3}_{\pm23.7}/\mathbf{53.2}_{\pm26.6}$ | $64.8_{\pm36.0}$ | $\mathbf{87.9}_{\pm10.2}/\mathbf{88.5}_{\pm6.0}$ | $12.2_{\pm7.0}$ |
| | GCN-EBM | $74.7_{\pm5.4}/73.2_{\pm5.7}$ | $92.0_{\pm0.8}$ | $66.5_{\pm5.9}/57.0_{\pm4.2}$ | $96.3_{\pm0.2}$ | $40.1_{\pm6.3}/34.5_{\pm6.4}$ | $90.2_{\pm1.9}$ | $64.5_{\pm4.4}/62.7_{\pm5.8}$ | $90.2_{\pm1.9}$ | $64.8_{\pm3.9}/63.2_{\pm4.4}$ | $90.2_{\pm1.9}$ | $40.7_{\pm6.3}/34.7_{\pm6.1}$ | $91.9_{\pm0.6}$ | $45.6_{\pm3.9}/37.2_{\pm3.6}$ | $97.7_{\pm0.6}$ |
| | GCN-HEAT | $74.7_{\pm5.4}/68.3_{\pm7.4}$ | $92.0_{\pm0.8}$ | $66.5_{\pm5.9}/50.0_{\pm8.1}$ | $96.3_{\pm0.2}$ | $40.1_{\pm6.3}/25.5_{\pm5.7}$ | $90.2_{\pm1.9}$ | $64.5_{\pm4.4}/61.1_{\pm6.3}$ | $90.2_{\pm1.9}$ | $64.8_{\pm3.9}/68.8_{\pm2.8}$ | $90.2_{\pm1.9}$ | $40.7_{\pm6.3}/22.6_{\pm2.8}$ | $91.9_{\pm0.6}$ | $45.6_{\pm3.9}/25.7_{\pm3.3}$ | $97.7_{\pm0.6}$ |
| | GCNSafe | $74.7_{\pm5.4}/69.0_{\pm7.8}$ | $92.0_{\pm0.8}$ | $66.5_{\pm5.9}/63.5_{\pm6.9}$ | $96.3_{\pm0.2}$ | $40.1_{\pm6.3}/32.2_{\pm5.8}$ | $90.2_{\pm1.9}$ | $64.5_{\pm4.4}/62.2_{\pm8.1}$ | $90.2_{\pm1.9}$ | $64.8_{\pm3.9}/62.3_{\pm3.3}$ | $90.2_{\pm1.9}$ | $40.7_{\pm6.3}/28.5_{\pm5.0}$ | $91.9_{\pm0.6}$ | $45.6_{\pm3.9}/29.0_{\pm3.1}$ | $97.7_{\pm0.6}$ |
| | **GCN-GEBM** | $74.7_{\pm5.4}/\mathbf{81.7}_{\pm6.6}$ | $92.0_{\pm0.8}$ | $66.5_{\pm5.9}/68.1_{\pm9.8}$ | $96.3_{\pm0.2}$ | $40.1_{\pm6.3}/17.3_{\pm3.4}$ | $90.2_{\pm1.9}$ | $64.5_{\pm4.4}/53.0_{\pm9.7}$ | $90.2_{\pm1.9}$ | $64.8_{\pm3.9}/64.2_{\pm5.0}$ | $90.2_{\pm1.9}$ | $40.7_{\pm6.3}/16.1_{\pm3.9}$ | $91.9_{\pm0.6}$ | $45.6_{\pm3.9}/18.3_{\pm2.3}$ | $97.7_{\pm0.6}$ |
| Coauthor-CS | GCN-MCD | $90.9_{\pm1.3}/59.4_{\pm1.3}$ | $93.1_{\pm0.3}$ | $77.0_{\pm3.0}/27.2_{\pm1.2}$ | $94.8_{\pm0.4}$ | $35.7_{\pm2.2}/13.9_{\pm1.0}$ | $92.1_{\pm0.3}$ | $46.6_{\pm4.6}/19.6_{\pm3.9}$ | $92.1_{\pm0.3}$ | $79.5_{\pm1.3}/79.5_{\pm1.8}$ | $92.1_{\pm0.3}$ | $30.2_{\pm2.1}/11.0_{\pm1.5}$ | $92.6_{\pm0.5}$ | $34.0_{\pm2.3}/10.6_{\pm1.0}$ | $98.8_{\pm0.4}$ |
| | GCN-DE | $90.3_{\pm1.1}/75.9_{\pm1.5}$ | $92.2_{\pm0.5}$ | $73.9_{\pm2.6}/47.2_{\pm2.3}$ | $94.0_{\pm0.5}$ | $34.8_{\pm2.1}/39.2_{\pm1.7}$ | $90.9_{\pm0.6}$ | $42.3_{\pm4.9}/55.1_{\pm3.3}$ | $90.9_{\pm0.6}$ | $78.2_{\pm0.9}/88.5_{\pm0.8}$ | $90.9_{\pm0.6}$ | $28.9_{\pm2.6}/22.2_{\pm1.8}$ | $90.9_{\pm0.6}$ | $29.7_{\pm2.2}/23.1_{\pm2.1}$ | $98.2_{\pm0.3}$ |
| | GCN-MCD+DE | $90.5_{\pm1.2}/75.6_{\pm1.8}$ | $92.2_{\pm0.4}$ | $73.2_{\pm3.0}/47.1_{\pm2.0}$ | $93.8_{\pm0.6}$ | $34.9_{\pm2.0}/40.2_{\pm1.6}$ | $90.8_{\pm0.6}$ | $41.7_{\pm4.8}/53.4_{\pm4.7}$ | $90.8_{\pm0.6}$ | $77.4_{\pm1.5}/88.5_{\pm1.1}$ | $90.8_{\pm0.6}$ | $28.9_{\pm2.4}/21.9_{\pm1.7}$ | $90.9_{\pm0.7}$ | $30.1_{\pm2.0}/23.2_{\pm1.3}$ | $98.2_{\pm0.3}$ |
| | GCN-BNN | $88.0_{\pm3.1}/86.5_{\pm3.5}$ | $91.9_{\pm0.5}$ | $72.5_{\pm4.1}/66.1_{\pm5.3}$ | $94.2_{\pm0.8}$ | $39.3_{\pm2.7}/35.7_{\pm3.6}$ | $91.2_{\pm0.5}$ | $55.5_{\pm8.0}/71.1_{\pm8.7}$ | $91.2_{\pm0.5}$ | $84.3_{\pm1.9}/90.0_{\pm1.2}$ | $91.2_{\pm0.5}$ | $38.7_{\pm2.8}/36.9_{\pm2.3}$ | $91.7_{\pm0.8}$ | $39.7_{\pm2.1}/38.1_{\pm2.7}$ | $98.8_{\pm0.5}$ |
| | GCN-Ens | $91.0_{\pm1.3}/86.2_{\pm2.1}$ | $93.1_{\pm0.3}$ | $77.0_{\pm2.8}/62.7_{\pm4.7}$ | $94.8_{\pm0.4}$ | $34.3_{\pm1.0}/18.6_{\pm1.9}$ | $92.2_{\pm0.2}$ | $44.4_{\pm4.4}/25.2_{\pm4.4}$ | $92.2_{\pm0.2}$ | $77.2_{\pm1.3}/84.8_{\pm1.3}$ | $92.2_{\pm0.2}$ | $30.2_{\pm2.0}/23.2_{\pm2.4}$ | $92.7_{\pm0.5}$ | $33.4_{\pm2.3}/27.4_{\pm1.8}$ | $98.9_{\pm0.4}$ |
| | GPN | $81.2_{\pm2.8}/91.0_{\pm1.3}$ | $86.7_{\pm0.9}$ | $67.8_{\pm4.6}/55.3_{\pm6.4}$ | $91.7_{\pm1.3}$ | $45.6_{\pm4.0}/24.3_{\pm2.2}$ | $85.4_{\pm1.7}$ | $46.1_{\pm4.1}/24.3_{\pm2.3}$ | $85.4_{\pm1.7}$ | $46.2_{\pm4.1}/24.4_{\pm2.3}$ | $85.4_{\pm1.7}$ | $45.8_{\pm3.2}/23.1_{\pm2.2}$ | $86.7_{\pm1.5}$ | $47.4_{\pm4.1}/23.7_{\pm2.6}$ | $94.8_{\pm1.2}$ |
| | SGCN | $90.9_{\pm4.1}/88.3_{\pm8.7}$ | $29.0_{\pm18.1}$ | $\mathbf{84.7}_{\pm8.5}/72.3_{\pm14.0}$ | $30.3_{\pm13.5}$ | $\mathbf{92.7}_{\pm4.0}/\mathbf{89.0}_{\pm8.1}$ | $14.0_{\pm8.5}$ | $\mathbf{92.8}_{\pm4.0}/\mathbf{92.2}_{\pm5.8}$ | $14.0_{\pm8.5}$ | $\mathbf{91.3}_{\pm2.3}/\mathbf{91.2}_{\pm3.6}$ | $14.0_{\pm8.5}$ | $\mathbf{89.1}_{\pm6.0}/\mathbf{85.6}_{\pm11.2}$ | $15.1_{\pm9.2}$ | $\mathbf{90.4}_{\pm7.9}/\mathbf{89.7}_{\pm8.7}$ | $13.1_{\pm9.3}$ |
| | GCN-EBM | $90.9_{\pm1.3}/90.4_{\pm1.3}$ | $93.1_{\pm0.4}$ | $76.7_{\pm2.8}/75.4_{\pm2.9}$ | $94.8_{\pm0.4}$ | $35.9_{\pm1.7}/32.4_{\pm2.2}$ | $92.0_{\pm0.2}$ | $48.2_{\pm3.7}/48.7_{\pm5.5}$ | $92.0_{\pm0.2}$ | $79.0_{\pm1.9}/75.9_{\pm2.3}$ | $92.0_{\pm0.2}$ | $30.1_{\pm1.7}/24.6_{\pm1.7}$ | $92.6_{\pm0.5}$ | $33.9_{\pm2.7}/27.4_{\pm2.5}$ | $98.9_{\pm0.4}$ |
| | GCN-HEAT | $90.9_{\pm1.3}/88.1_{\pm2.5}$ | $93.1_{\pm0.4}$ | $76.7_{\pm2.8}/59.8_{\pm7.9}$ | $94.8_{\pm0.4}$ | $35.9_{\pm1.7}/26.4_{\pm3.0}$ | $92.0_{\pm0.2}$ | $48.2_{\pm3.7}/32.8_{\pm5.1}$ | $92.0_{\pm0.2}$ | $79.0_{\pm1.9}/86.8_{\pm1.7}$ | $92.0_{\pm0.2}$ | $30.1_{\pm1.7}/20.6_{\pm2.5}$ | $92.6_{\pm0.5}$ | $33.9_{\pm2.7}/24.9_{\pm2.3}$ | $98.9_{\pm0.4}$ |
| | GCNSafe | $90.9_{\pm1.3}/\mathbf{93.1}_{\pm0.7}$ | $93.1_{\pm0.4}$ | $76.7_{\pm2.8}/\mathbf{78.6}_{\pm2.7}$ | $94.8_{\pm0.4}$ | $35.9_{\pm1.7}/28.8_{\pm1.9}$ | $92.0_{\pm0.2}$ | $48.2_{\pm3.7}/43.7_{\pm5.0}$ | $92.0_{\pm0.2}$ | $79.0_{\pm1.9}/76.5_{\pm2.3}$ | $92.0_{\pm0.2}$ | $30.1_{\pm1.7}/19.6_{\pm1.0}$ | $92.6_{\pm0.5}$ | $33.9_{\pm2.7}/19.0_{\pm1.8}$ | $98.9_{\pm0.4}$ |
| | **GCN-GEBM** | $90.9_{\pm1.3}/\mathbf{93.7}_{\pm0.8}$ | $93.1_{\pm0.4}$ | $76.7_{\pm2.8}/74.9_{\pm5.3}$ | $94.8_{\pm0.4}$ | $35.9_{\pm1.7}/24.5_{\pm2.9}$ | $92.0_{\pm0.2}$ | $48.2_{\pm3.7}/33.9_{\pm7.5}$ | $92.0_{\pm0.2}$ | $79.0_{\pm1.9}/85.4_{\pm1.8}$ | $92.0_{\pm0.2}$ | $30.1_{\pm1.7}/17.2_{\pm3.1}$ | $92.6_{\pm0.5}$ | $33.9_{\pm2.7}/19.0_{\pm2.0}$ | $98.9_{\pm0.4}$ |
| Coauthor-Physics | GCN-MCD | $77.8_{\pm2.2}/21.3_{\pm1.0}$ | $96.6_{\pm1.2}$ | $87.9_{\pm3.7}/26.3_{\pm0.6}$ | $98.1_{\pm0.3}$ | $39.0_{\pm2.7}/10.9_{\pm1.4}$ | $92.9_{\pm0.7}$ | $37.8_{\pm5.6}/12.9_{\pm3.2}$ | $92.9_{\pm0.7}$ | $59.5_{\pm1.3}/54.5_{\pm2.4}$ | $92.9_{\pm0.7}$ | $34.1_{\pm1.6}/7.7_{\pm1.0}$ | $94.5_{\pm0.8}$ | $39.3_{\pm3.2}/6.5_{\pm0.7}$ | $98.8_{\pm0.2}$ |
| | GCN-DE | $77.8_{\pm2.2}/57.9_{\pm1.4}$ | $96.5_{\pm1.0}$ | $86.8_{\pm3.2}/71.5_{\pm2.5}$ | $97.9_{\pm0.4}$ | $39.5_{\pm2.9}/38.3_{\pm1.9}$ | $92.3_{\pm0.6}$ | $36.6_{\pm4.9}/37.8_{\pm3.9}$ | $92.3_{\pm0.6}$ | $61.8_{\pm1.3}/69.9_{\pm1.3}$ | $92.3_{\pm0.6}$ | $33.9_{\pm2.3}/25.0_{\pm1.4}$ | $94.1_{\pm0.7}$ | $37.4_{\pm2.1}/26.6_{\pm1.7}$ | $98.4_{\pm0.3}$ |
| | GCN-MCD+DE | $77.2_{\pm2.7}/57.1_{\pm1.8}$ | $96.5_{\pm1.0}$ | $87.2_{\pm3.2}/72.1_{\pm2.1}$ | $97.9_{\pm0.3}$ | $39.1_{\pm3.5}/37.6_{\pm2.1}$ | $92.3_{\pm0.7}$ | $37.7_{\pm6.3}/38.7_{\pm4.5}$ | $92.3_{\pm0.7}$ | $62.1_{\pm1.6}/70.4_{\pm1.2}$ | $92.3_{\pm0.7}$ | $33.6_{\pm2.8}/25.0_{\pm1.8}$ | $94.2_{\pm0.7}$ | $36.6_{\pm2.5}/25.9_{\pm2.3}$ | $98.4_{\pm0.3}$ |
| | GCN-BNN | $73.5_{\pm3.5}/70.0_{\pm4.2}$ | $96.6_{\pm1.0}$ | $79.6_{\pm6.6}/81.3_{\pm6.5}$ | $97.9_{\pm0.3}$ | $\mathbf{42.3}_{\pm4.4}/37.2_{\pm5.6}$ | $92.4_{\pm0.9}$ | $\mathbf{55.0}_{\pm12.5}/\mathbf{65.3}_{\pm15.3}$ | $92.4_{\pm0.9}$ | $\mathbf{72.4}_{\pm12.3}/\mathbf{77.0}_{\pm6.1}$ | $92.4_{\pm0.9}$ | $\mathbf{37.6}_{\pm3.1}/37.5_{\pm5.0}$ | $94.2_{\pm1.0}$ | $\mathbf{40.2}_{\pm3.2}/38.2_{\pm2.8}$ | $98.9_{\pm0.3}$ |
| | GCN-Ens | $77.8_{\pm2.2}/61.3_{\pm6.1}$ | $96.7_{\pm1.2}$ | $88.1_{\pm3.7}/77.0_{\pm5.0}$ | $98.1_{\pm0.3}$ | $39.6_{\pm2.4}/17.6_{\pm2.9}$ | $92.9_{\pm0.7}$ | $37.5_{\pm5.4}/14.7_{\pm1.8}$ | $92.9_{\pm0.7}$ | $59.2_{\pm1.0}/63.3_{\pm1.1}$ | $92.9_{\pm0.7}$ | $34.3_{\pm1.8}/22.7_{\pm3.1}$ | $94.5_{\pm0.8}$ | $39.2_{\pm3.4}/30.2_{\pm1.6}$ | $98.7_{\pm0.3}$ |
| | GPN | $70.3_{\pm3.9}/74.7_{\pm3.9}$ | $96.8_{\pm0.5}$ | $66.9_{\pm4.8}/87.1_{\pm2.8}$ | $97.7_{\pm0.3}$ | $\mathbf{42.1}_{\pm2.7}/21.2_{\pm4.3}$ | $92.3_{\pm0.7}$ | $40.6_{\pm2.1}/18.7_{\pm2.3}$ | $92.3_{\pm0.7}$ | $40.8_{\pm2.1}/18.7_{\pm2.3}$ | $92.3_{\pm0.7}$ | $\mathbf{40.1}_{\pm1.9}/18.5_{\pm2.6}$ | $93.3_{\pm0.8}$ | $\mathbf{40.9}_{\pm2.3}/20.1_{\pm2.4}$ | $98.1_{\pm0.4}$ |
| | SGCN | $78.0_{\pm3.0}/76.0_{\pm6.5}$ | $96.5_{\pm1.0}$ | $88.5_{\pm2.6}/88.1_{\pm3.2}$ | $98.1_{\pm0.4}$ | $\mathbf{42.1}_{\pm4.1}/30.2_{\pm9.5}$ | $92.4_{\pm0.9}$ | $43.1_{\pm5.7}/27.6_{\pm9.3}$ | $92.4_{\pm0.9}$ | $58.7_{\pm2.7}/53.3_{\pm4.4}$ | $92.4_{\pm0.9}$ | $33.8_{\pm2.7}/23.8_{\pm5.3}$ | $94.0_{\pm0.7}$ | $37.8_{\pm2.5}/28.3_{\pm5.2}$ | $98.6_{\pm0.4}$ |
| | GCN-EBM | $77.7_{\pm2.1}/78.4_{\pm3.1}$ | $96.7_{\pm1.1}$ | $88.0_{\pm3.7}/89.2_{\pm3.1}$ | $98.1_{\pm0.3}$ | $38.8_{\pm2.1}/34.4_{\pm3.3}$ | $92.8_{\pm0.7}$ | $38.2_{\pm5.6}/33.9_{\pm6.6}$ | $92.8_{\pm0.7}$ | $59.2_{\pm1.6}/55.9_{\pm2.3}$ | $92.8_{\pm0.7}$ | $33.9_{\pm1.6}/28.4_{\pm1.7}$ | $94.5_{\pm0.7}$ | $39.5_{\pm3.0}/33.7_{\pm3.5}$ | $98.7_{\pm0.2}$ |
| | GCN-HEAT | $77.7_{\pm2.1}/63.2_{\pm4.9}$ | $96.7_{\pm1.1}$ | $88.0_{\pm3.7}/77.0_{\pm5.9}$ | $98.1_{\pm0.3}$ | $38.8_{\pm2.1}/26.6_{\pm3.3}$ | $92.8_{\pm0.7}$ | $38.2_{\pm5.6}/25.6_{\pm4.1}$ | $92.8_{\pm0.7}$ | $59.2_{\pm1.6}/64.0_{\pm3.4}$ | $92.8_{\pm0.7}$ | $33.9_{\pm1.6}/26.4_{\pm3.6}$ | $94.5_{\pm0.7}$ | $39.5_{\pm3.0}/24.8_{\pm2.2}$ | $98.7_{\pm0.2}$ |
| | GCNSafe | $77.7_{\pm2.1}/\mathbf{84.5}_{\pm1.6}$ | $96.7_{\pm1.1}$ | $88.0_{\pm3.7}/\mathbf{92.4}_{\pm2.3}$ | $98.1_{\pm0.3}$ | $38.8_{\pm2.1}/36.2_{\pm3.3}$ | $92.8_{\pm0.7}$ | $38.2_{\pm5.6}/35.3_{\pm7.1}$ | $92.8_{\pm0.7}$ | $59.2_{\pm1.6}/58.0_{\pm2.4}$ | $92.8_{\pm0.7}$ | $33.9_{\pm1.6}/27.1_{\pm2.9}$ | $94.5_{\pm0.7}$ | $39.5_{\pm3.0}/28.1_{\pm2.7}$ | $98.7_{\pm0.2}$ |
| | **GCN-GEBM** | $77.7_{\pm2.1}/\mathbf{83.4}_{\pm3.0}$ | $96.7_{\pm1.1}$ | $88.0_{\pm3.7}/\mathbf{92.6}_{\pm2.7}$ | $98.1_{\pm0.3}$ | $38.8_{\pm2.1}/31.0_{\pm5.4}$ | $92.8_{\pm0.7}$ | $38.2_{\pm5.6}/32.0_{\pm8.7}$ | $92.8_{\pm0.7}$ | $59.2_{\pm1.6}/61.1_{\pm4.8}$ | $92.8_{\pm0.7}$ | $33.9_{\pm1.6}/32.3_{\pm3.1}$ | $94.5_{\pm0.7}$ | $39.5_{\pm3.0}/34.4_{\pm3.2}$ | $98.7_{\pm0.2}$ |

Table 19: Misclassification detection AUC-PR (↑) using aleatoric or epistemic uncertainty (best and runner-up) in a transductive setting.

| Dataset | Model | LoC (last) Clean / I.d. / O.o.D. | LoC (heterophilic) Clean / I.d. / O.o.D. | Ber($\hat{p}$) (near) Clean / I.d. / O.o.D. | Ber(0.5) Clean / I.d. / O.o.D. | $\mathcal{N}(0,1)$ (far) Clean / I.d. / O.o.D. | Page Rank Clean / I.d. / O.o.D. | Homophily Clean / I.d. / O.o.D. |
|---|---|---|---|---|---|---|---|---|
| CoraML | BGCN | $6.4_{\pm1.2}$ / $\mathbf{5.7}_{\pm1.2}$ / $69.7_{\pm2.6}$ | $7.5_{\pm2.0}$ / $\mathbf{7.1}_{\pm2.2}$ / $68.0_{\pm4.6}$ | $12.2_{\pm2.6}$ / $\mathbf{10.6}_{\pm5.4}$ / $\mathbf{12.1}_{\pm5.2}$ | $12.2_{\pm2.6}$ / $\mathbf{10.6}_{\pm3.7}$ / $19.3_{\pm8.2}$ | $12.2_{\pm2.6}$ / $48.5_{\pm5.4}$ / $56.7_{\pm4.1}$ | $11.8_{\pm2.5}$ / $11.2_{\pm1.9}$ / $\mathbf{8.6}_{\pm2.5}$ | $\mathbf{4.9}_{\pm1.2}$ / $\mathbf{5.8}_{\pm1.3}$ / $\mathbf{9.4}_{\pm2.0}$ |
| | GCN-MCD | $\mathbf{6.1}_{\pm1.2}$ / $7.7_{\pm1.5}$ / $51.5_{\pm2.6}$ | $\mathbf{7.0}_{\pm2.0}$ / $8.7_{\pm1.4}$ / $49.8_{\pm2.4}$ | $\mathbf{7.7}_{\pm1.7}$ / $27.1_{\pm2.4}$ / $30.5_{\pm2.7}$ | $\mathbf{7.7}_{\pm1.7}$ / $23.7_{\pm3.0}$ / $21.3_{\pm5.3}$ | $\mathbf{7.7}_{\pm1.7}$ / $42.8_{\pm3.1}$ / $65.2_{\pm1.3}$ | $\mathbf{6.8}_{\pm1.4}$ / $\mathbf{6.5}_{\pm1.0}$ / $15.7_{\pm2.1}$ | $12.0_{\pm1.2}$ / $21.0_{\pm1.0}$ / $15.1_{\pm1.9}$ |
| | GCN-DE | $11.3_{\pm2.3}$ / $13.0_{\pm2.3}$ / $\mathbf{47.2}_{\pm1.5}$ | $11.7_{\pm2.0}$ / $12.5_{\pm2.1}$ / $46.4_{\pm1.8}$ | $16.7_{\pm2.8}$ / $28.4_{\pm3.4}$ / $24.0_{\pm2.8}$ | $16.7_{\pm2.8}$ / $21.8_{\pm4.8}$ / $\mathbf{12.9}_{\pm4.6}$ | $16.7_{\pm2.8}$ / $34.0_{\pm2.9}$ / $57.3_{\pm2.5}$ | $12.4_{\pm1.7}$ / $12.0_{\pm2.1}$ / $19.4_{\pm2.6}$ | $22.3_{\pm2.0}$ / $29.9_{\pm2.3}$ / $21.1_{\pm3.5}$ |
| | GCN-Ens | $\mathbf{6.0}_{\pm1.2}$ / $7.6_{\pm1.6}$ / $51.3_{\pm2.5}$ | $\mathbf{7.0}_{\pm1.5}$ / $\mathbf{8.3}_{\pm1.1}$ / $50.0_{\pm2.1}$ | $\mathbf{7.2}_{\pm1.7}$ / $26.3_{\pm2.0}$ / $31.7_{\pm1.7}$ | $\mathbf{7.2}_{\pm1.7}$ / $24.3_{\pm3.4}$ / $21.5_{\pm6.2}$ | $\mathbf{7.2}_{\pm1.7}$ / $40.4_{\pm1.4}$ / $59.4_{\pm1.6}$ | $\mathbf{6.5}_{\pm1.1}$ / $\mathbf{6.9}_{\pm0.9}$ / $15.9_{\pm2.1}$ | $12.4_{\pm1.0}$ / $21.6_{\pm0.8}$ / $15.4_{\pm1.8}$ |
| | GCN-MCD+DE | $11.4_{\pm1.7}$ / $12.8_{\pm1.9}$ / $\mathbf{47.2}_{\pm1.5}$ | $12.3_{\pm2.0}$ / $13.1_{\pm2.3}$ / $\mathbf{46.0}_{\pm1.7}$ | $17.1_{\pm2.6}$ / $29.4_{\pm3.3}$ / $23.7_{\pm3.0}$ | $17.1_{\pm2.6}$ / $22.5_{\pm4.4}$ / $\mathbf{13.4}_{\pm4.0}$ | $17.1_{\pm2.6}$ / $37.8_{\pm3.3}$ / $56.5_{\pm2.2}$ | $12.1_{\pm2.3}$ / $12.3_{\pm2.4}$ / $20.2_{\pm2.7}$ | $22.4_{\pm2.9}$ / $29.9_{\pm3.2}$ / $21.7_{\pm3.8}$ |
| | SGCN | $32.9_{\pm2.1}$ / $34.9_{\pm1.9}$ / $\mathbf{34.0}_{\pm0.8}$ | $31.5_{\pm2.8}$ / $33.2_{\pm2.6}$ / $\mathbf{34.9}_{\pm0.9}$ | $18.8_{\pm16.3}$ / $18.1_{\pm20.1}$ / $\mathbf{15.9}_{\pm18.2}$ | $18.8_{\pm16.3}$ / $16.1_{\pm19.8}$ / $\mathbf{13.4}_{\pm17.3}$ | $18.8_{\pm16.3}$ / $\mathbf{6.4}_{\pm6.1}$ / $\mathbf{10.5}_{\pm14.4}$ | $39.0_{\pm13.0}$ / $38.3_{\pm12.6}$ / $44.8_{\pm15.8}$ | $54.7_{\pm1.3}$ / $63.0_{\pm1.6}$ / $46.2_{\pm2.7}$ |
| | GPN | $\mathbf{6.0}_{\pm1.0}$ / $\mathbf{7.4}_{\pm1.3}$ / $54.8_{\pm2.1}$ | $9.6_{\pm1.8}$ / $10.9_{\pm2.7}$ / $52.0_{\pm4.0}$ | $9.7_{\pm1.8}$ / $\mathbf{15.1}_{\pm4.6}$ / $17.5_{\pm5.2}$ | $9.7_{\pm1.8}$ / $\mathbf{14.9}_{\pm4.4}$ / $15.9_{\pm5.3}$ | $9.7_{\pm1.8}$ / $\mathbf{12.1}_{\pm2.9}$ / $\mathbf{13.3}_{\pm2.6}$ | $12.9_{\pm2.5}$ / $12.7_{\pm2.3}$ / $\mathbf{13.0}_{\pm2.3}$ | $\mathbf{5.1}_{\pm0.8}$ / $\mathbf{9.6}_{\pm2.0}$ / $15.9_{\pm6.5}$ |
| | GCN (Ours) | $6.2_{\pm1.3}$ / $7.8_{\pm1.8}$ / $51.4_{\pm2.7}$ | $\mathbf{6.9}_{\pm1.4}$ / $\mathbf{8.3}_{\pm1.1}$ / $50.2_{\pm2.1}$ | $7.8_{\pm1.7}$ / $26.4_{\pm2.5}$ / $31.5_{\pm3.1}$ | $7.8_{\pm1.7}$ / $24.4_{\pm3.5}$ / $21.1_{\pm5.5}$ | $7.8_{\pm1.7}$ / $45.0_{\pm2.7}$ / $64.8_{\pm2.2}$ | $7.0_{\pm1.3}$ / $\mathbf{6.9}_{\pm0.7}$ / $16.3_{\pm2.4}$ | $12.1_{\pm1.3}$ / $21.2_{\pm0.9}$ / $\mathbf{14.7}_{\pm2.5}$ |
| CoraML LLM | BGCN | $4.8_{\pm1.3}$ / $4.8_{\pm0.9}$ / $69.3_{\pm4.1}$ | $4.6_{\pm1.7}$ / $\mathbf{4.8}_{\pm1.6}$ / $64.2_{\pm8.9}$ | n.a. / n.a. / n.a. | $7.4_{\pm1.5}$ / $\mathbf{19.1}_{\pm6.2}$ / $30.9_{\pm4.6}$ | $7.4_{\pm1.5}$ / $16.4_{\pm3.5}$ / $39.0_{\pm5.2}$ | $\mathbf{5.8}_{\pm1.4}$ / $6.5_{\pm1.3}$ / $\mathbf{7.6}_{\pm2.0}$ | $4.6_{\pm0.6}$ / $\mathbf{3.5}_{\pm0.6}$ / $13.4_{\pm2.4}$ |
| | GCN-MCD | $\mathbf{4.0}_{\pm0.7}$ / $\mathbf{4.2}_{\pm0.8}$ / $62.8_{\pm2.0}$ | $\mathbf{4.1}_{\pm1.0}$ / $\mathbf{5.3}_{\pm0.8}$ / $60.3_{\pm6.3}$ | n.a. / n.a. / n.a. | $\mathbf{6.3}_{\pm1.2}$ / $22.1_{\pm2.6}$ / $32.9_{\pm1.9}$ | $\mathbf{6.3}_{\pm1.2}$ / $15.9_{\pm2.6}$ / $41.6_{\pm5.7}$ | $6.3_{\pm1.0}$ / $\mathbf{5.4}_{\pm1.0}$ / $9.8_{\pm1.2}$ | $\mathbf{4.4}_{\pm0.9}$ / $8.9_{\pm1.3}$ / $\mathbf{6.9}_{\pm1.0}$ |
| | GCN-DE | $5.4_{\pm1.3}$ / $5.3_{\pm1.3}$ / $61.6_{\pm2.6}$ | $5.6_{\pm1.2}$ / $6.3_{\pm1.4}$ / $58.3_{\pm3.3}$ | n.a. / n.a. / n.a. | $8.0_{\pm2.0}$ / $21.5_{\pm3.1}$ / $\mathbf{25.1}_{\pm2.7}$ | $8.0_{\pm2.0}$ / $16.1_{\pm1.9}$ / $42.8_{\pm3.0}$ | $7.8_{\pm1.4}$ / $6.7_{\pm1.5}$ / $9.9_{\pm2.0}$ | $8.3_{\pm1.4}$ / $12.2_{\pm1.8}$ / $8.5_{\pm1.9}$ |
| | GCN-Ens | $4.1_{\pm0.6}$ / $\mathbf{4.3}_{\pm0.8}$ / $62.7_{\pm2.2}$ | $\mathbf{4.0}_{\pm0.7}$ / $5.5_{\pm0.8}$ / $60.0_{\pm5.8}$ | n.a. / n.a. / n.a. | $6.4_{\pm1.3}$ / $22.8_{\pm2.2}$ / $34.0_{\pm0.7}$ | $6.4_{\pm1.3}$ / $\mathbf{13.2}_{\pm2.3}$ / $38.3_{\pm3.1}$ | $6.7_{\pm0.9}$ / $\mathbf{5.4}_{\pm1.0}$ / $9.8_{\pm2.0}$ | $4.6_{\pm0.6}$ / $8.9_{\pm1.2}$ / $7.4_{\pm1.2}$ |
| | GCN-MCD+DE | $5.3_{\pm0.9}$ / $5.5_{\pm1.2}$ / $61.4_{\pm2.0}$ | $5.6_{\pm1.3}$ / $6.1_{\pm1.2}$ / $58.1_{\pm5.4}$ | n.a. / n.a. / n.a. | $8.0_{\pm1.8}$ / $21.6_{\pm3.9}$ / $26.3_{\pm2.6}$ | $8.0_{\pm1.8}$ / $16.3_{\pm3.1}$ / $41.5_{\pm4.1}$ | $7.6_{\pm1.6}$ / $7.2_{\pm1.6}$ / $10.6_{\pm2.5}$ | $7.8_{\pm1.4}$ / $11.1_{\pm1.7}$ / $7.9_{\pm1.3}$ |
| | SGCN | $23.3_{\pm1.8}$ / $24.5_{\pm1.7}$ / $\mathbf{40.6}_{\pm1.5}$ | $27.3_{\pm2.1}$ / $28.5_{\pm2.1}$ / $\mathbf{39.5}_{\pm2.7}$ | n.a. / n.a. / n.a. | $35.3_{\pm11.0}$ / $47.0_{\pm16.6}$ / $46.5_{\pm17.1}$ | $35.3_{\pm11.0}$ / $19.6_{\pm7.0}$ / $\mathbf{12.2}_{\pm4.7}$ | $38.3_{\pm7.1}$ / $35.4_{\pm6.2}$ / $43.3_{\pm8.2}$ | $40.5_{\pm10.1}$ / $48.0_{\pm12.8}$ / $33.1_{\pm9.8}$ |
| | GPN | $5.5_{\pm1.1}$ / $6.3_{\pm1.3}$ / $\mathbf{59.4}_{\pm3.1}$ | $7.8_{\pm2.4}$ / $8.8_{\pm2.6}$ / $\mathbf{54.2}_{\pm4.3}$ | n.a. / n.a. / n.a. | $7.0_{\pm1.2}$ / $\mathbf{12.7}_{\pm3.4}$ / $\mathbf{15.4}_{\pm4.0}$ | $7.0_{\pm1.2}$ / $\mathbf{10.4}_{\pm2.3}$ / $\mathbf{11.1}_{\pm2.5}$ | $9.5_{\pm1.3}$ / $8.9_{\pm1.3}$ / $\mathbf{8.9}_{\pm1.2}$ | $\mathbf{4.3}_{\pm0.8}$ / $\mathbf{8.5}_{\pm1.6}$ / $15.0_{\pm5.9}$ |
| | GCN (Ours) | $\mathbf{3.9}_{\pm0.8}$ / $\mathbf{4.2}_{\pm1.0}$ / $63.0_{\pm2.3}$ | $\mathbf{4.1}_{\pm0.8}$ / $\mathbf{5.3}_{\pm0.8}$ / $60.0_{\pm5.9}$ | n.a. / n.a. / n.a. | $\mathbf{6.2}_{\pm1.1}$ / $23.1_{\pm3.0}$ / $33.0_{\pm2.1}$ | $\mathbf{6.2}_{\pm1.1}$ / $16.5_{\pm2.2}$ / $41.0_{\pm3.1}$ | $\mathbf{6.1}_{\pm0.8}$ / $\mathbf{5.6}_{\pm0.9}$ / $9.8_{\pm1.6}$ | $4.5_{\pm0.9}$ / $8.9_{\pm1.5}$ / $\mathbf{7.3}_{\pm1.0}$ |
| Citeseer | BGCN | $7.5_{\pm0.9}$ / $7.3_{\pm0.9}$ / $65.3_{\pm2.4}$ | $\mathbf{4.1}_{\pm0.8}$ / $\mathbf{4.1}_{\pm0.7}$ / $66.0_{\pm2.7}$ | $10.8_{\pm1.9}$ / $\mathbf{16.0}_{\pm8.3}$ / $12.8_{\pm3.9}$ | $10.8_{\pm1.9}$ / $\mathbf{11.7}_{\pm3.6}$ / $30.2_{\pm13.2}$ | $10.8_{\pm1.9}$ / $35.3_{\pm3.0}$ / $58.8_{\pm4.7}$ | $10.0_{\pm0.9}$ / $8.6_{\pm1.0}$ / $\mathbf{4.9}_{\pm0.7}$ | $8.0_{\pm0.9}$ / $\mathbf{4.6}_{\pm0.8}$ / $\mathbf{5.8}_{\pm1.0}$ |
| | GCN-MCD | $\mathbf{4.2}_{\pm0.8}$ / $4.2_{\pm0.8}$ / $54.5_{\pm2.3}$ | $4.2_{\pm0.7}$ / $4.4_{\pm0.7}$ / $55.1_{\pm1.7}$ | $\mathbf{7.1}_{\pm1.4}$ / $18.7_{\pm2.3}$ / $13.9_{\pm2.3}$ | $\mathbf{7.1}_{\pm1.4}$ / $12.3_{\pm2.7}$ / $13.2_{\pm5.9}$ | $\mathbf{7.1}_{\pm1.4}$ / $34.3_{\pm2.9}$ / $63.9_{\pm2.7}$ | $\mathbf{5.0}_{\pm0.9}$ / $\mathbf{4.9}_{\pm0.8}$ / $13.1_{\pm0.6}$ | $6.4_{\pm1.1}$ / $12.8_{\pm2.0}$ / $10.4_{\pm2.6}$ |
| | GCN-DE | $7.3_{\pm1.8}$ / $7.4_{\pm1.8}$ / $51.8_{\pm2.7}$ | $8.7_{\pm1.4}$ / $8.8_{\pm1.6}$ / $51.9_{\pm1.7}$ | $10.9_{\pm1.8}$ / $16.7_{\pm2.7}$ / $\mathbf{7.6}_{\pm1.8}$ | $10.9_{\pm1.8}$ / $13.9_{\pm3.2}$ / $12.0_{\pm3.3}$ | $10.9_{\pm1.8}$ / $25.4_{\pm2.6}$ / $57.4_{\pm2.4}$ | $8.4_{\pm1.5}$ / $9.1_{\pm1.9}$ / $15.2_{\pm1.8}$ | $10.3_{\pm2.0}$ / $16.2_{\pm2.4}$ / $13.8_{\pm2.6}$ |
| | GCN-Ens | $\mathbf{4.1}_{\pm0.7}$ / $\mathbf{4.1}_{\pm0.7}$ / $54.2_{\pm2.1}$ | $4.4_{\pm0.8}$ / $4.6_{\pm0.9}$ / $54.9_{\pm1.6}$ | $\mathbf{7.1}_{\pm1.4}$ / $19.8_{\pm2.8}$ / $15.0_{\pm1.9}$ | $\mathbf{7.1}_{\pm1.4}$ / $15.8_{\pm2.8}$ / $\mathbf{10.6}_{\pm2.5}$ | $\mathbf{7.1}_{\pm1.4}$ / $32.3_{\pm1.3}$ / $59.5_{\pm1.2}$ | $\mathbf{5.2}_{\pm0.8}$ / $\mathbf{4.7}_{\pm0.7}$ / $13.0_{\pm0.5}$ | $6.8_{\pm1.0}$ / $12.9_{\pm2.0}$ / $10.8_{\pm2.8}$ |
| | GCN-MCD+DE | $7.2_{\pm1.7}$ / $6.9_{\pm1.4}$ / $\mathbf{51.5}_{\pm2.8}$ | $9.1_{\pm1.9}$ / $8.9_{\pm1.8}$ / $\mathbf{51.7}_{\pm1.9}$ | $10.9_{\pm2.1}$ / $16.8_{\pm2.5}$ / $\mathbf{8.6}_{\pm2.2}$ | $10.9_{\pm2.1}$ / $15.0_{\pm4.1}$ / $12.0_{\pm5.8}$ | $10.9_{\pm2.1}$ / $25.1_{\pm2.5}$ / $57.1_{\pm2.1}$ | $8.1_{\pm1.5}$ / $8.7_{\pm1.9}$ / $15.6_{\pm2.4}$ | $10.4_{\pm1.7}$ / $15.5_{\pm2.5}$ / $13.6_{\pm2.4}$ |
| | SGCN | $27.1_{\pm1.7}$ / $27.4_{\pm1.7}$ / $\mathbf{35.9}_{\pm1.3}$ | $31.5_{\pm1.2}$ / $31.9_{\pm1.3}$ / $\mathbf{35.4}_{\pm0.8}$ | $37.0_{\pm1.8}$ / $45.2_{\pm2.4}$ / $32.2_{\pm2.2}$ | $37.0_{\pm1.8}$ / $43.9_{\pm2.7}$ / $28.6_{\pm5.8}$ | $37.0_{\pm1.8}$ / $\mathbf{21.2}_{\pm2.6}$ / $\mathbf{46.3}_{\pm2.0}$ | $35.7_{\pm1.8}$ / $35.9_{\pm1.8}$ / $48.7_{\pm2.0}$ | $36.0_{\pm1.4}$ / $47.3_{\pm1.5}$ / $45.3_{\pm1.8}$ |
| | GPN | $4.4_{\pm0.9}$ / $4.4_{\pm0.9}$ / $57.6_{\pm4.1}$ | $\mathbf{3.4}_{\pm0.6}$ / $\mathbf{3.4}_{\pm0.6}$ / $62.1_{\pm3.3}$ | $\mathbf{6.1}_{\pm1.1}$ / $\mathbf{11.6}_{\pm2.7}$ / $9.8_{\pm2.6}$ | $\mathbf{6.1}_{\pm1.1}$ / $\mathbf{11.0}_{\pm8.1}$ / $12.8_{\pm11.9}$ | $\mathbf{6.1}_{\pm1.1}$ / $\mathbf{7.2}_{\pm1.5}$ / $\mathbf{6.8}_{\pm1.4}$ | $6.3_{\pm1.4}$ / $6.1_{\pm1.3}$ / $\mathbf{6.6}_{\pm1.7}$ | $\mathbf{5.6}_{\pm1.4}$ / $\mathbf{6.3}_{\pm1.9}$ / $\mathbf{6.2}_{\pm1.6}$ |
| | GCN (Ours) | $\mathbf{4.1}_{\pm0.9}$ / $\mathbf{4.0}_{\pm0.9}$ / $54.4_{\pm2.2}$ | $4.4_{\pm0.9}$ / $4.6_{\pm0.9}$ / $54.9_{\pm1.6}$ | $7.3_{\pm1.6}$ / $19.3_{\pm3.0}$ / $13.7_{\pm2.4}$ | $7.3_{\pm1.6}$ / $13.9_{\pm3.3}$ / $\mathbf{11.3}_{\pm6.6}$ | $7.3_{\pm1.6}$ / $35.9_{\pm2.8}$ / $64.0_{\pm1.3}$ | $5.4_{\pm1.0}$ / $5.1_{\pm0.7}$ / $12.8_{\pm0.7}$ | $\mathbf{6.1}_{\pm1.0}$ / $12.8_{\pm1.8}$ / $10.7_{\pm2.6}$ |
| PubMed | BGCN | $4.8_{\pm1.2}$ / $5.0_{\pm1.2}$ / $93.6_{\pm1.5}$ | $8.9_{\pm5.3}$ / $8.6_{\pm5.3}$ / $81.9_{\pm6.2}$ | n.a. / n.a. / n.a. | $4.3_{\pm1.3}$ / $\mathbf{8.3}_{\pm4.2}$ / $12.0_{\pm8.3}$ | $4.3_{\pm1.3}$ / $27.1_{\pm2.8}$ / $47.9_{\pm2.1}$ | $11.9_{\pm5.7}$ / $12.8_{\pm5.5}$ / $9.5_{\pm5.5}$ | $9.4_{\pm1.8}$ / $\mathbf{4.3}_{\pm1.9}$ / $19.7_{\pm2.1}$ |
| | GCN-MCD | $\mathbf{2.0}_{\pm0.8}$ / $\mathbf{2.6}_{\pm1.1}$ / $83.3_{\pm3.8}$ | $2.7_{\pm1.2}$ / $\mathbf{2.8}_{\pm1.6}$ / $71.4_{\pm1.6}$ | n.a. / n.a. / n.a. | $4.0_{\pm0.5}$ / $12.9_{\pm3.3}$ / $\mathbf{7.7}_{\pm2.6}$ | $4.0_{\pm0.5}$ / $23.4_{\pm0.9}$ / $46.9_{\pm1.1}$ | $\mathbf{2.6}_{\pm0.8}$ / $\mathbf{3.1}_{\pm0.7}$ / $6.7_{\pm2.0}$ | $2.9_{\pm0.5}$ / $10.9_{\pm1.5}$ / $6.0_{\pm1.5}$ |
| | GCN-DE | $2.7_{\pm0.8}$ / $3.3_{\pm1.1}$ / $82.7_{\pm3.2}$ | $2.6_{\pm1.2}$ / $\mathbf{2.8}_{\pm1.2}$ / $\mathbf{70.5}_{\pm2.2}$ | n.a. / n.a. / n.a. | $5.4_{\pm1.2}$ / $\mathbf{9.4}_{\pm3.1}$ / $8.0_{\pm5.5}$ | $5.4_{\pm1.2}$ / $21.7_{\pm2.0}$ / $48.7_{\pm2.0}$ | $4.1_{\pm1.3}$ / $3.5_{\pm1.1}$ / $\mathbf{6.0}_{\pm1.7}$ | $5.3_{\pm1.7}$ / $11.0_{\pm2.0}$ / $\mathbf{4.6}_{\pm1.3}$ |
| | GCN-Ens | $\mathbf{1.8}_{\pm0.6}$ / $\mathbf{2.5}_{\pm1.0}$ / $83.5_{\pm3.8}$ | $2.7_{\pm1.0}$ / $\mathbf{2.8}_{\pm1.4}$ / $71.3_{\pm1.2}$ | n.a. / n.a. / n.a. | $\mathbf{3.9}_{\pm0.4}$ / $13.1_{\pm3.3}$ / $8.2_{\pm2.4}$ | $\mathbf{3.9}_{\pm0.4}$ / $22.7_{\pm1.0}$ / $45.5_{\pm1.4}$ | $\mathbf{2.7}_{\pm0.4}$ / $\mathbf{3.1}_{\pm1.0}$ / $6.7_{\pm1.8}$ | $\mathbf{2.8}_{\pm0.6}$ / $11.1_{\pm1.0}$ / $5.8_{\pm1.2}$ |
| | GCN-MCD+DE | $2.6_{\pm1.1}$ / $3.0_{\pm1.2}$ / $82.9_{\pm3.8}$ | $\mathbf{2.5}_{\pm0.9}$ / $\mathbf{2.8}_{\pm1.0}$ / $70.7_{\pm1.8}$ | n.a. / n.a. / n.a. | $5.9_{\pm1.8}$ / $10.4_{\pm4.1}$ / $10.3_{\pm8.3}$ | $5.9_{\pm1.8}$ / $20.7_{\pm2.6}$ / $47.8_{\pm2.2}$ | $4.4_{\pm1.8}$ / $4.0_{\pm1.2}$ / $6.5_{\pm2.0}$ | $5.1_{\pm1.9}$ / $10.5_{\pm2.2}$ / $\mathbf{4.7}_{\pm1.4}$ |
| | SGCN | $9.8_{\pm0.8}$ / $11.0_{\pm0.8}$ / $\mathbf{75.3}_{\pm2.1}$ | $8.5_{\pm1.4}$ / $9.1_{\pm1.4}$ / $\mathbf{67.1}_{\pm1.8}$ | n.a. / n.a. / n.a. | $16.5_{\pm1.2}$ / $25.7_{\pm2.4}$ / $14.9_{\pm4.6}$ | $16.5_{\pm1.2}$ / $\mathbf{10.8}_{\pm1.7}$ / $\mathbf{35.3}_{\pm2.1}$ | $16.4_{\pm1.6}$ / $13.6_{\pm1.4}$ / $23.3_{\pm1.8}$ | $19.1_{\pm2.1}$ / $30.6_{\pm2.3}$ / $12.6_{\pm2.0}$ |
| | GPN | $3.5_{\pm1.3}$ / $5.1_{\pm1.4}$ / $\mathbf{82.1}_{\pm2.8}$ | $\mathbf{2.0}_{\pm0.9}$ / $\mathbf{2.4}_{\pm1.2}$ / $74.3_{\pm2.8}$ | n.a. / n.a. / n.a. | $\mathbf{3.8}_{\pm1.8}$ / $12.0_{\pm2.4}$ / $9.1_{\pm3.8}$ | $\mathbf{3.8}_{\pm1.8}$ / $\mathbf{8.3}_{\pm2.9}$ / $\mathbf{8.4}_{\pm3.6}$ | $4.1_{\pm1.5}$ / $3.7_{\pm1.5}$ / $\mathbf{3.8}_{\pm1.6}$ | $3.8_{\pm1.2}$ / $\mathbf{5.0}_{\pm2.0}$ / $14.4_{\pm4.7}$ |
| | GCN (Ours) | $\mathbf{2.0}_{\pm0.8}$ / $\mathbf{2.5}_{\pm1.1}$ / $83.4_{\pm3.9}$ | $2.7_{\pm0.9}$ / $\mathbf{2.8}_{\pm1.2}$ / $71.4_{\pm1.2}$ | n.a. / n.a. / n.a. | $\mathbf{3.9}_{\pm0.5}$ / $12.8_{\pm3.5}$ / $\mathbf{7.6}_{\pm2.8}$ | $\mathbf{3.9}_{\pm0.5}$ / $22.4_{\pm1.5}$ / $46.5_{\pm1.4}$ | $2.8_{\pm0.7}$ / $\mathbf{3.3}_{\pm0.9}$ / $6.8_{\pm2.1}$ | $\mathbf{2.7}_{\pm0.7}$ / $11.1_{\pm1.1}$ / $6.1_{\pm1.4}$ |
| Amazon Photo | BGCN | $4.0_{\pm1.3}$ / $\mathbf{3.9}_{\pm1.4}$ / $86.7_{\pm2.8}$ | $\mathbf{2.5}_{\pm1.4}$ / $\mathbf{2.4}_{\pm1.2}$ / $82.6_{\pm6.0}$ | $\mathbf{4.4}_{\pm1.0}$ / $\mathbf{8.7}_{\pm3.3}$ / $\mathbf{9.4}_{\pm4.2}$ | $\mathbf{4.4}_{\pm1.0}$ / $23.7_{\pm10.8}$ / $30.0_{\pm12.3}$ | $\mathbf{4.4}_{\pm1.0}$ / $31.5_{\pm4.0}$ / $39.4_{\pm3.2}$ | $\mathbf{4.3}_{\pm1.4}$ / $3.8_{\pm1.0}$ / $\mathbf{4.5}_{\pm1.4}$ | $\mathbf{1.6}_{\pm0.3}$ / $\mathbf{2.8}_{\pm0.8}$ / $\mathbf{6.6}_{\pm1.6}$ |
| | GCN-MCD | $\mathbf{3.8}_{\pm0.6}$ / $4.9_{\pm0.7}$ / $73.1_{\pm2.4}$ | $4.4_{\pm1.5}$ / $5.8_{\pm1.6}$ / $70.9_{\pm6.4}$ | $6.4_{\pm1.0}$ / $25.9_{\pm1.7}$ / $26.4_{\pm2.7}$ | $6.4_{\pm1.0}$ / $\mathbf{11.3}_{\pm3.8}$ / $12.4_{\pm3.8}$ | $6.4_{\pm1.0}$ / $22.9_{\pm3.0}$ / $35.1_{\pm2.7}$ | $\mathbf{4.5}_{\pm1.4}$ / $\mathbf{3.0}_{\pm0.8}$ / $13.4_{\pm2.2}$ | $4.6_{\pm0.4}$ / $13.1_{\pm0.7}$ / $10.1_{\pm1.2}$ |
| | GCN-DE | $5.4_{\pm0.8}$ / $6.7_{\pm1.0}$ / $69.4_{\pm2.0}$ | $5.8_{\pm1.0}$ / $7.1_{\pm1.1}$ / $67.7_{\pm4.9}$ | $9.0_{\pm1.4}$ / $29.1_{\pm1.8}$ / $28.6_{\pm2.1}$ | $9.0_{\pm1.4}$ / $13.7_{\pm5.7}$ / $\mathbf{11.0}_{\pm4.0}$ | $9.0_{\pm1.4}$ / $20.7_{\pm3.3}$ / $32.6_{\pm3.7}$ | $6.4_{\pm1.4}$ / $4.3_{\pm0.7}$ / $14.9_{\pm1.6}$ | $9.1_{\pm1.1}$ / $17.3_{\pm1.6}$ / $14.3_{\pm1.9}$ |
| | GCN-Ens | $\mathbf{3.7}_{\pm0.6}$ / $4.9_{\pm0.4}$ / $73.0_{\pm2.0}$ | $4.2_{\pm0.9}$ / $5.6_{\pm1.1}$ / $71.2_{\pm5.9}$ | $6.4_{\pm0.4}$ / $26.0_{\pm0.6}$ / $27.1_{\pm2.3}$ | $6.4_{\pm0.4}$ / $11.6_{\pm3.5}$ / $12.5_{\pm3.3}$ | $6.4_{\pm0.4}$ / $19.2_{\pm4.0}$ / $30.8_{\pm3.3}$ | $4.6_{\pm1.1}$ / $\mathbf{3.0}_{\pm0.6}$ / $13.5_{\pm1.2}$ | $4.7_{\pm0.3}$ / $13.1_{\pm0.6}$ / $10.1_{\pm1.1}$ |
| | GCN-MCD+DE | $5.4_{\pm1.1}$ / $6.5_{\pm1.1}$ / $69.4_{\pm2.1}$ | $5.9_{\pm0.9}$ / $7.3_{\pm1.0}$ / $67.8_{\pm5.6}$ | $9.8_{\pm1.7}$ / $29.8_{\pm1.9}$ / $28.6_{\pm2.0}$ | $9.8_{\pm1.7}$ / $13.6_{\pm6.4}$ / $11.1_{\pm3.6}$ | $9.8_{\pm1.7}$ / $20.5_{\pm4.7}$ / $32.6_{\pm4.2}$ | $7.1_{\pm1.9}$ / $4.8_{\pm1.5}$ / $15.5_{\pm2.1}$ | $8.8_{\pm1.0}$ / $17.1_{\pm1.4}$ / $13.7_{\pm1.5}$ |
| | SGCN | $32.8_{\pm2.0}$ / $35.0_{\pm1.9}$ / $\mathbf{39.6}_{\pm1.2}$ | $32.4_{\pm2.0}$ / $34.8_{\pm1.9}$ / $\mathbf{41.4}_{\pm2.5}$ | $7.0_{\pm7.5}$ / $\mathbf{7.1}_{\pm7.2}$ / $\mathbf{6.8}_{\pm6.7}$ | $7.0_{\pm7.5}$ / $\mathbf{6.3}_{\pm6.5}$ / $\mathbf{6.2}_{\pm6.0}$ | $7.0_{\pm7.5}$ / $\mathbf{6.2}_{\pm5.1}$ / $\mathbf{5.4}_{\pm4.7}$ | $7.5_{\pm4.4}$ / $7.4_{\pm4.5}$ / $\mathbf{7.4}_{\pm3.7}$ | $4.8_{\pm4.2}$ / $\mathbf{5.0}_{\pm4.2}$ / $\mathbf{5.9}_{\pm4.6}$ |
| | GPN | $11.9_{\pm1.9}$ / $11.8_{\pm2.2}$ / $\mathbf{61.2}_{\pm3.6}$ | $4.7_{\pm1.5}$ / $6.1_{\pm2.0}$ / $\mathbf{66.0}_{\pm7.0}$ | $16.4_{\pm2.3}$ / $19.5_{\pm1.7}$ / $20.4_{\pm2.0}$ | $16.4_{\pm2.3}$ / $19.4_{\pm1.6}$ / $20.6_{\pm2.2}$ | $16.4_{\pm2.3}$ / $\mathbf{17.9}_{\pm1.7}$ / $\mathbf{18.8}_{\pm2.0}$ | $19.4_{\pm1.8}$ / $19.5_{\pm1.9}$ / $16.9_{\pm2.0}$ | $\mathbf{1.9}_{\pm0.3}$ / $15.5_{\pm2.9}$ / $11.4_{\pm3.1}$ |
| | GCN (Ours) | $\mathbf{3.7}_{\pm0.6}$ / $\mathbf{4.8}_{\pm0.7}$ / $73.4_{\pm2.3}$ | $\mathbf{4.1}_{\pm1.1}$ / $\mathbf{5.5}_{\pm1.2}$ / $71.6_{\pm5.9}$ | $\mathbf{6.3}_{\pm0.9}$ / $25.8_{\pm1.3}$ / $26.5_{\pm2.2}$ | $\mathbf{6.3}_{\pm0.9}$ / $11.4_{\pm3.6}$ / $11.9_{\pm4.0}$ | $\mathbf{6.3}_{\pm0.9}$ / $19.6_{\pm3.0}$ / $30.9_{\pm3.2}$ | $4.6_{\pm1.3}$ / $\mathbf{3.1}_{\pm0.7}$ / $13.6_{\pm1.9}$ | $4.5_{\pm0.4}$ / $13.0_{\pm0.7}$ / $9.8_{\pm1.4}$ |
| Amazon Computers | BGCN | $\mathbf{6.8}_{\pm6.0}$ / $\mathbf{6.6}_{\pm5.4}$ / $74.5_{\pm5.1}$ | $\mathbf{2.7}_{\pm0.5}$ / $\mathbf{2.6}_{\pm0.7}$ / $73.0_{\pm8.3}$ | $\mathbf{6.4}_{\pm1.9}$ / $\mathbf{12.5}_{\pm4.5}$ / $\mathbf{12.8}_{\pm4.7}$ | $\mathbf{6.4}_{\pm1.9}$ / $\mathbf{15.1}_{\pm9.1}$ / $\mathbf{15.0}_{\pm12.0}$ | $\mathbf{6.4}_{\pm1.9}$ / $27.9_{\pm7.1}$ / $38.9_{\pm5.7}$ | $\mathbf{6.2}_{\pm2.1}$ / $7.3_{\pm2.3}$ / $\mathbf{7.8}_{\pm3.2}$ | $\mathbf{3.3}_{\pm0.9}$ / $\mathbf{3.6}_{\pm1.0}$ / $13.6_{\pm2.7}$ |
| | GCN-MCD | $8.4_{\pm2.3}$ / $8.9_{\pm2.3}$ / $64.0_{\pm5.1}$ | $\mathbf{5.9}_{\pm1.9}$ / $8.0_{\pm2.5}$ / $56.8_{\pm9.2}$ | $\mathbf{8.5}_{\pm1.8}$ / $29.7_{\pm1.9}$ / $30.4_{\pm1.7}$ | $\mathbf{8.5}_{\pm1.8}$ / $22.4_{\pm3.2}$ / $19.6_{\pm2.8}$ | $\mathbf{8.5}_{\pm1.8}$ / $16.3_{\pm2.2}$ / $27.0_{\pm3.4}$ | $\mathbf{7.2}_{\pm1.9}$ / $\mathbf{4.4}_{\pm1.4}$ / $19.7_{\pm4.1}$ | $5.7_{\pm0.9}$ / $13.3_{\pm1.2}$ / $\mathbf{7.4}_{\pm1.5}$ |
| | GCN-DE | $9.9_{\pm2.2}$ / $10.7_{\pm2.2}$ / $60.1_{\pm4.3}$ | $7.8_{\pm1.8}$ / $9.9_{\pm2.3}$ / $\mathbf{54.1}_{\pm7.6}$ | $13.1_{\pm3.0}$ / $33.9_{\pm2.5}$ / $33.3_{\pm2.4}$ | $13.1_{\pm3.0}$ / $23.0_{\pm8.4}$ / $20.5_{\pm6.2}$ | $13.1_{\pm3.0}$ / $18.4_{\pm3.2}$ / $28.7_{\pm2.9}$ | $9.7_{\pm2.0}$ / $6.0_{\pm1.4}$ / $20.4_{\pm3.5}$ | $9.8_{\pm1.4}$ / $16.4_{\pm1.9}$ / $9.7_{\pm1.9}$ |
| | GCN-Ens | $\mathbf{8.3}_{\pm2.3}$ / $\mathbf{8.8}_{\pm2.4}$ / $64.1_{\pm4.9}$ | $\mathbf{5.9}_{\pm1.3}$ / $\mathbf{7.9}_{\pm1.9}$ / $56.8_{\pm9.2}$ | $9.1_{\pm1.5}$ / $30.2_{\pm0.6}$ / $30.8_{\pm1.6}$ | $9.1_{\pm1.5}$ / $22.7_{\pm3.5}$ / $20.2_{\pm3.2}$ | $9.1_{\pm1.5}$ / $\mathbf{13.7}_{\pm2.9}$ / $23.4_{\pm2.4}$ | $7.6_{\pm1.6}$ / $4.6_{\pm0.8}$ / $20.3_{\pm3.7}$ | $5.7_{\pm0.6}$ / $13.4_{\pm1.2}$ / $\mathbf{7.4}_{\pm1.6}$ |
| | GCN-MCD+DE | $10.2_{\pm2.4}$ / $10.7_{\pm2.3}$ / $60.2_{\pm3.4}$ | $8.1_{\pm2.2}$ / $10.3_{\pm2.8}$ / $54.6_{\pm9.0}$ | $12.2_{\pm2.7}$ / $33.0_{\pm1.8}$ / $32.5_{\pm2.6}$ | $12.2_{\pm2.7}$ / $22.7_{\pm8.3}$ / $19.6_{\pm6.8}$ | $12.2_{\pm2.7}$ / $18.0_{\pm3.8}$ / $28.0_{\pm3.6}$ | $9.2_{\pm2.4}$ / $5.7_{\pm1.7}$ / $19.5_{\pm3.6}$ | $10.2_{\pm1.3}$ / $16.9_{\pm1.8}$ / $9.5_{\pm1.9}$ |
| | SGCN | $24.9_{\pm13.4}$ / $25.0_{\pm13.5}$ / $\mathbf{19.4}_{\pm5.0}$ | $33.6_{\pm14.0}$ / $34.2_{\pm14.2}$ / $\mathbf{21.2}_{\pm7.3}$ | $8.7_{\pm8.5}$ / $\mathbf{8.7}_{\pm8.4}$ / $\mathbf{8.9}_{\pm8.1}$ | $8.7_{\pm8.5}$ / $\mathbf{8.7}_{\pm8.4}$ / $\mathbf{8.9}_{\pm8.1}$ | $8.7_{\pm8.5}$ / $\mathbf{6.7}_{\pm7.4}$ / $\mathbf{6.3}_{\pm6.8}$ | $13.6_{\pm10.6}$ / $13.4_{\pm10.5}$ / $\mathbf{14.1}_{\pm10.3}$ | $7.3_{\pm6.0}$ / $\mathbf{6.9}_{\pm6.4}$ / $\mathbf{7.2}_{\pm6.1}$ |
| | GPN | $18.6_{\pm3.8}$ / $18.7_{\pm3.8}$ / $\mathbf{48.1}_{\pm5.9}$ | $8.7_{\pm1.8}$ / $10.9_{\pm2.4}$ / $63.5_{\pm9.4}$ | $19.2_{\pm3.8}$ / $22.8_{\pm4.2}$ / $24.3_{\pm4.5}$ | $19.2_{\pm3.8}$ / $22.2_{\pm4.1}$ / $24.2_{\pm4.7}$ | $19.2_{\pm3.8}$ / $20.7_{\pm3.9}$ / $\mathbf{22.5}_{\pm4.5}$ | $19.1_{\pm3.2}$ / $18.3_{\pm2.7}$ / $18.7_{\pm2.6}$ | $\mathbf{3.5}_{\pm0.9}$ / $12.2_{\pm3.5}$ / $9.0_{\pm3.7}$ |
| | GCN (Ours) | $\mathbf{8.3}_{\pm2.7}$ / $\mathbf{8.8}_{\pm2.7}$ / $63.7_{\pm5.0}$ | $6.4_{\pm1.8}$ / $8.5_{\pm2.4}$ / $56.6_{\pm9.6}$ | $9.0_{\pm2.2}$ / $30.2_{\pm1.8}$ / $30.8_{\pm1.7}$ | $9.0_{\pm2.2}$ / $22.9_{\pm4.5}$ / $20.2_{\pm3.5}$ | $9.0_{\pm2.2}$ / $16.3_{\pm3.1}$ / $28.2_{\pm3.3}$ | $7.4_{\pm1.8}$ / $\mathbf{4.3}_{\pm1.2}$ / $19.9_{\pm4.0}$ | $5.5_{\pm0.9}$ / $13.3_{\pm1.5}$ / $7.7_{\pm1.8}$ |
| Coauthor Physics | BGCN | $\mathbf{0.8}_{\pm0.3}$ / $\mathbf{0.8}_{\pm0.3}$ / $77.8_{\pm3.4}$ | $\mathbf{0.7}_{\pm0.3}$ / $\mathbf{0.8}_{\pm0.5}$ / $79.3_{\pm5.7}$ | $\mathbf{2.4}_{\pm1.0}$ / $\mathbf{6.1}_{\pm2.1}$ / $\mathbf{8.0}_{\pm2.5}$ | $\mathbf{2.4}_{\pm1.0}$ / $\mathbf{11.8}_{\pm4.2}$ / $\mathbf{13.0}_{\pm6.4}$ | $\mathbf{2.4}_{\pm1.0}$ / $18.3_{\pm8.9}$ / $22.4_{\pm7.2}$ | $2.4_{\pm0.6}$ / $\mathbf{2.0}_{\pm0.7}$ / $\mathbf{1.5}_{\pm0.4}$ | $\mathbf{1.0}_{\pm0.4}$ / $\mathbf{1.3}_{\pm0.4}$ / $3.5_{\pm1.0}$ |
| | GCN-MCD | $\mathbf{2.6}_{\pm0.7}$ / $\mathbf{2.9}_{\pm0.7}$ / $67.7_{\pm1.5}$ | $\mathbf{1.9}_{\pm0.3}$ / $\mathbf{2.2}_{\pm0.3}$ / $66.5_{\pm2.1}$ | $3.8_{\pm0.6}$ / $18.2_{\pm2.0}$ / $19.7_{\pm3.2}$ | $3.8_{\pm0.6}$ / $21.5_{\pm1.6}$ / $26.4_{\pm1.4}$ | $3.8_{\pm0.6}$ / $21.3_{\pm3.3}$ / $36.7_{\pm2.8}$ | $\mathbf{2.3}_{\pm0.6}$ / $\mathbf{1.8}_{\pm0.5}$ / $\mathbf{8.6}_{\pm1.1}$ | $4.0_{\pm0.4}$ / $\mathbf{6.2}_{\pm0.5}$ / $\mathbf{3.0}_{\pm0.6}$ |
| | GCN-DE | $3.5_{\pm0.7}$ / $3.8_{\pm0.8}$ / $\mathbf{65.8}_{\pm1.5}$ | $2.7_{\pm0.5}$ / $3.0_{\pm0.4}$ / $65.1_{\pm2.4}$ | $6.4_{\pm1.2}$ / $20.8_{\pm2.8}$ / $19.8_{\pm4.1}$ | $6.4_{\pm1.2}$ / $24.6_{\pm1.8}$ / $27.5_{\pm3.0}$ | $6.4_{\pm1.2}$ / $25.5_{\pm3.7}$ / $39.2_{\pm3.2}$ | $3.5_{\pm0.7}$ / $3.0_{\pm0.6}$ / $9.5_{\pm1.1}$ | $6.8_{\pm0.7}$ / $8.7_{\pm0.8}$ / $5.3_{\pm0.9}$ |
| | GCN-Ens | $\mathbf{2.6}_{\pm0.5}$ / $\mathbf{2.9}_{\pm0.5}$ / $67.7_{\pm1.4}$ | $\mathbf{1.9}_{\pm0.3}$ / $\mathbf{2.2}_{\pm0.3}$ / $66.6_{\pm1.8}$ | $\mathbf{3.7}_{\pm0.8}$ / $18.1_{\pm2.1}$ / $19.5_{\pm3.6}$ | $\mathbf{3.7}_{\pm0.8}$ / $21.6_{\pm1.8}$ / $26.5_{\pm1.0}$ | $\mathbf{3.7}_{\pm0.8}$ / $19.3_{\pm3.3}$ / $34.6_{\pm2.8}$ | $\mathbf{2.2}_{\pm0.4}$ / $\mathbf{1.8}_{\pm0.4}$ / $\mathbf{8.6}_{\pm0.9}$ | $3.9_{\pm0.3}$ / $\mathbf{6.2}_{\pm0.4}$ / $\mathbf{3.1}_{\pm0.5}$ |
| | GCN-MCD+DE | $3.3_{\pm0.6}$ / $3.5_{\pm0.7}$ / $66.0_{\pm1.6}$ | $2.7_{\pm0.4}$ / $3.0_{\pm0.4}$ / $\mathbf{65.0}_{\pm2.6}$ | $6.5_{\pm1.3}$ / $20.2_{\pm2.8}$ / $18.7_{\pm3.7}$ | $6.5_{\pm1.3}$ / $24.4_{\pm2.3}$ / $27.1_{\pm4.0}$ | $6.5_{\pm1.3}$ / $24.7_{\pm3.0}$ / $38.4_{\pm2.1}$ | $3.4_{\pm0.6}$ / $2.9_{\pm0.6}$ / $9.3_{\pm1.1}$ | $7.0_{\pm0.7}$ / $8.9_{\pm0.9}$ / $5.7_{\pm1.0}$ |
| | SGCN | $24.6_{\pm1.7}$ / $25.1_{\pm1.7}$ / $\mathbf{47.1}_{\pm1.3}$ | $24.1_{\pm1.7}$ / $24.9_{\pm1.7}$ / $\mathbf{44.6}_{\pm1.1}$ | $38.8_{\pm1.8}$ / $55.1_{\pm1.9}$ / $54.1_{\pm2.7}$ | $38.8_{\pm1.8}$ / $54.5_{\pm2.8}$ / $53.4_{\pm5.4}$ | $38.8_{\pm1.8}$ / $\mathbf{13.5}_{\pm2.1}$ / $\mathbf{14.4}_{\pm1.8}$ | $37.0_{\pm2.2}$ / $34.6_{\pm2.3}$ / $48.2_{\pm1.9}$ | $41.2_{\pm1.7}$ / $45.5_{\pm1.6}$ / $39.9_{\pm1.0}$ |
| | GPN | $8.2_{\pm0.7}$ / $8.8_{\pm0.7}$ / $66.7_{\pm2.3}$ | $5.3_{\pm0.7}$ / $5.8_{\pm0.8}$ / $79.0_{\pm2.7}$ | $8.4_{\pm1.3}$ / $\mathbf{15.2}_{\pm1.9}$ / $\mathbf{17.8}_{\pm2.3}$ | $8.4_{\pm1.3}$ / $\mathbf{12.8}_{\pm1.4}$ / $\mathbf{14.3}_{\pm1.6}$ | $8.4_{\pm1.3}$ / $\mathbf{12.4}_{\pm1.4}$ / $\mathbf{13.6}_{\pm1.4}$ | $13.0_{\pm1.3}$ / $13.1_{\pm1.4}$ / $12.1_{\pm1.7}$ | $\mathbf{0.4}_{\pm0.1}$ / $10.0_{\pm6.7}$ / $9.7_{\pm8.9}$ |
| | GCN (Ours) | $\mathbf{2.6}_{\pm0.7}$ / $\mathbf{2.9}_{\pm0.7}$ / $67.6_{\pm1.6}$ | $2.0_{\pm0.3}$ / $\mathbf{2.2}_{\pm0.3}$ / $66.5_{\pm1.6}$ | $\mathbf{3.7}_{\pm1.0}$ / $17.9_{\pm2.3}$ / $19.2_{\pm3.6}$ | $\mathbf{3.7}_{\pm1.0}$ / $21.4_{\pm2.1}$ / $26.5_{\pm1.4}$ | $\mathbf{3.7}_{\pm1.0}$ / $20.2_{\pm2.4}$ / $35.4_{\pm3.0}$ | $\mathbf{2.3}_{\pm0.6}$ / $\mathbf{1.8}_{\pm0.5}$ / $\mathbf{8.6}_{\pm1.4}$ | $4.0_{\pm0.4}$ / $\mathbf{6.2}_{\pm0.5}$ / $\mathbf{3.1}_{\pm0.5}$ |
| Coauthor-CS | BGCN | $\mathbf{3.0}_{\pm0.5}$ / $\mathbf{2.5}_{\pm0.5}$ / $67.0_{\pm3.1}$ | $\mathbf{1.4}_{\pm0.3}$ / $\mathbf{1.2}_{\pm0.3}$ / $69.6_{\pm2.5}$ | $\mathbf{3.7}_{\pm0.6}$ / $\mathbf{10.7}_{\pm2.4}$ / $\mathbf{17.7}_{\pm2.6}$ | $\mathbf{3.7}_{\pm0.6}$ / $\mathbf{14.9}_{\pm5.2}$ / $\mathbf{15.2}_{\pm4.8}$ | $\mathbf{3.7}_{\pm0.6}$ / $25.5_{\pm7.4}$ / $33.5_{\pm8.7}$ | $\mathbf{3.8}_{\pm0.5}$ / $\mathbf{3.2}_{\pm0.5}$ / $\mathbf{2.1}_{\pm0.7}$ | $\mathbf{1.3}_{\pm0.3}$ / $\mathbf{2.3}_{\pm0.4}$ / $\mathbf{4.0}_{\pm0.8}$ |
| | GCN-MCD | $\mathbf{5.4}_{\pm0.9}$ / $\mathbf{7.3}_{\pm1.1}$ / $47.5_{\pm2.2}$ | $\mathbf{4.2}_{\pm0.4}$ / $4.9_{\pm0.5}$ / $52.1_{\pm2.1}$ | $6.0_{\pm0.9}$ / $31.5_{\pm1.4}$ / $40.2_{\pm1.7}$ | $6.0_{\pm0.9}$ / $32.6_{\pm1.9}$ / $37.6_{\pm2.5}$ | $6.0_{\pm0.9}$ / $30.9_{\pm1.8}$ / $54.1_{\pm1.3}$ | $4.5_{\pm0.5}$ / $\mathbf{3.3}_{\pm0.5}$ / $12.3_{\pm1.2}$ | $5.3_{\pm0.4}$ / $11.5_{\pm0.6}$ / $9.6_{\pm0.6}$ |
| | GCN-DE | $9.2_{\pm1.2}$ / $11.3_{\pm1.4}$ / $\mathbf{43.3}_{\pm2.4}$ | $7.4_{\pm0.8}$ / $8.1_{\pm0.9}$ / $47.9_{\pm1.8}$ | $12.9_{\pm1.6}$ / $36.5_{\pm1.9}$ / $39.3_{\pm1.7}$ | $12.9_{\pm1.6}$ / $36.0_{\pm2.5}$ / $31.7_{\pm4.5}$ | $12.9_{\pm1.6}$ / $24.5_{\pm1.6}$ / $43.1_{\pm1.6}$ | $8.4_{\pm1.3}$ / $7.1_{\pm1.3}$ / $16.7_{\pm2.1}$ | $12.6_{\pm1.2}$ / $17.4_{\pm1.4}$ / $14.7_{\pm1.5}$ |
| | GCN-Ens | $5.5_{\pm0.6}$ / $7.4_{\pm1.0}$ / $47.1_{\pm1.8}$ | $\mathbf{4.2}_{\pm0.3}$ / $\mathbf{4.8}_{\pm0.2}$ / $52.2_{\pm1.8}$ | $6.1_{\pm0.9}$ / $31.7_{\pm1.3}$ / $40.6_{\pm1.2}$ | $6.1_{\pm0.9}$ / $33.4_{\pm2.0}$ / $39.2_{\pm1.9}$ | $6.1_{\pm0.9}$ / $23.6_{\pm1.2}$ / $45.1_{\pm0.9}$ | $\mathbf{4.3}_{\pm0.5}$ / $\mathbf{3.2}_{\pm0.3}$ / $12.1_{\pm1.1}$ | $5.5_{\pm0.4}$ / $11.7_{\pm0.5}$ / $10.3_{\pm0.6}$ |
| | GCN-MCD+DE | $9.1_{\pm1.2}$ / $10.9_{\pm1.3}$ / $43.9_{\pm2.2}$ | $7.4_{\pm0.8}$ / $8.0_{\pm0.8}$ / $48.6_{\pm2.0}$ | $12.8_{\pm1.4}$ / $36.3_{\pm1.7}$ / $39.5_{\pm1.7}$ | $12.8_{\pm1.4}$ / $36.2_{\pm2.3}$ / $32.9_{\pm4.0}$ | $12.8_{\pm1.4}$ / $26.2_{\pm1.6}$ / $44.4_{\pm1.4}$ | $8.1_{\pm1.1}$ / $6.7_{\pm1.0}$ / $16.1_{\pm1.6}$ | $12.3_{\pm0.8}$ / $17.1_{\pm1.0}$ / $14.7_{\pm1.1}$ |
| | SGCN | $20.6_{\pm14.6}$ / $19.8_{\pm14.4}$ / $\mathbf{10.1}_{\pm0.0}$ | $22.0_{\pm15.0}$ / $21.4_{\pm14.8}$ / $\mathbf{10.1}_{\pm0.0}$ | $6.6_{\pm4.8}$ / $\mathbf{4.4}_{\pm4.1}$ / $\mathbf{4.3}_{\pm3.7}$ | $6.6_{\pm4.8}$ / $\mathbf{4.4}_{\pm4.3}$ / $\mathbf{4.4}_{\pm4.1}$ | $6.6_{\pm4.8}$ / $\mathbf{2.6}_{\pm1.8}$ / $\mathbf{2.1}_{\pm1.3}$ | $7.0_{\pm7.8}$ / $7.1_{\pm7.9}$ / $\mathbf{7.1}_{\pm8.2}$ | $7.7_{\pm6.1}$ / $\mathbf{6.3}_{\pm5.5}$ / $\mathbf{4.2}_{\pm4.4}$ |
| | GPN | $18.0_{\pm2.2}$ / $19.5_{\pm2.5}$ / $49.3_{\pm3.7}$ | $19.0_{\pm1.6}$ / $20.3_{\pm1.7}$ / $\mathbf{46.2}_{\pm2.4}$ | $14.7_{\pm2.7}$ / $24.6_{\pm3.2}$ / $26.2_{\pm4.0}$ | $14.7_{\pm2.7}$ / $21.5_{\pm2.6}$ / $20.9_{\pm3.1}$ | $14.7_{\pm2.7}$ / $\mathbf{21.0}_{\pm2.6}$ / $\mathbf{20.1}_{\pm2.9}$ | $21.4_{\pm2.3}$ / $22.4_{\pm2.1}$ / $20.5_{\pm2.1}$ | $\mathbf{1.6}_{\pm0.3}$ / $9.9_{\pm4.8}$ / $10.6_{\pm5.8}$ |
| | GCN (Ours) | $5.5_{\pm1.0}$ / $\mathbf{7.3}_{\pm1.2}$ / $47.3_{\pm2.0}$ | $4.3_{\pm0.4}$ / $5.0_{\pm0.4}$ / $51.9_{\pm2.1}$ | $\mathbf{5.9}_{\pm1.0}$ / $31.6_{\pm1.5}$ / $40.0_{\pm1.3}$ | $\mathbf{5.9}_{\pm1.0}$ / $32.5_{\pm2.0}$ / $36.6_{\pm3.1}$ | $\mathbf{5.9}_{\pm1.0}$ / $30.6_{\pm2.5}$ / $54.0_{\pm1.4}$ | $4.5_{\pm0.5}$ / $\mathbf{3.2}_{\pm0.5}$ / $12.2_{\pm1.4}$ | $5.5_{\pm0.5}$ / $11.7_{\pm0.6}$ / $9.9_{\pm0.6}$ |

Table 20: ECE (↓) for different backbones on clean data as well as i.d. and o.o.d. nodes after a distribution shift in an inductive setting (best and runner-up).

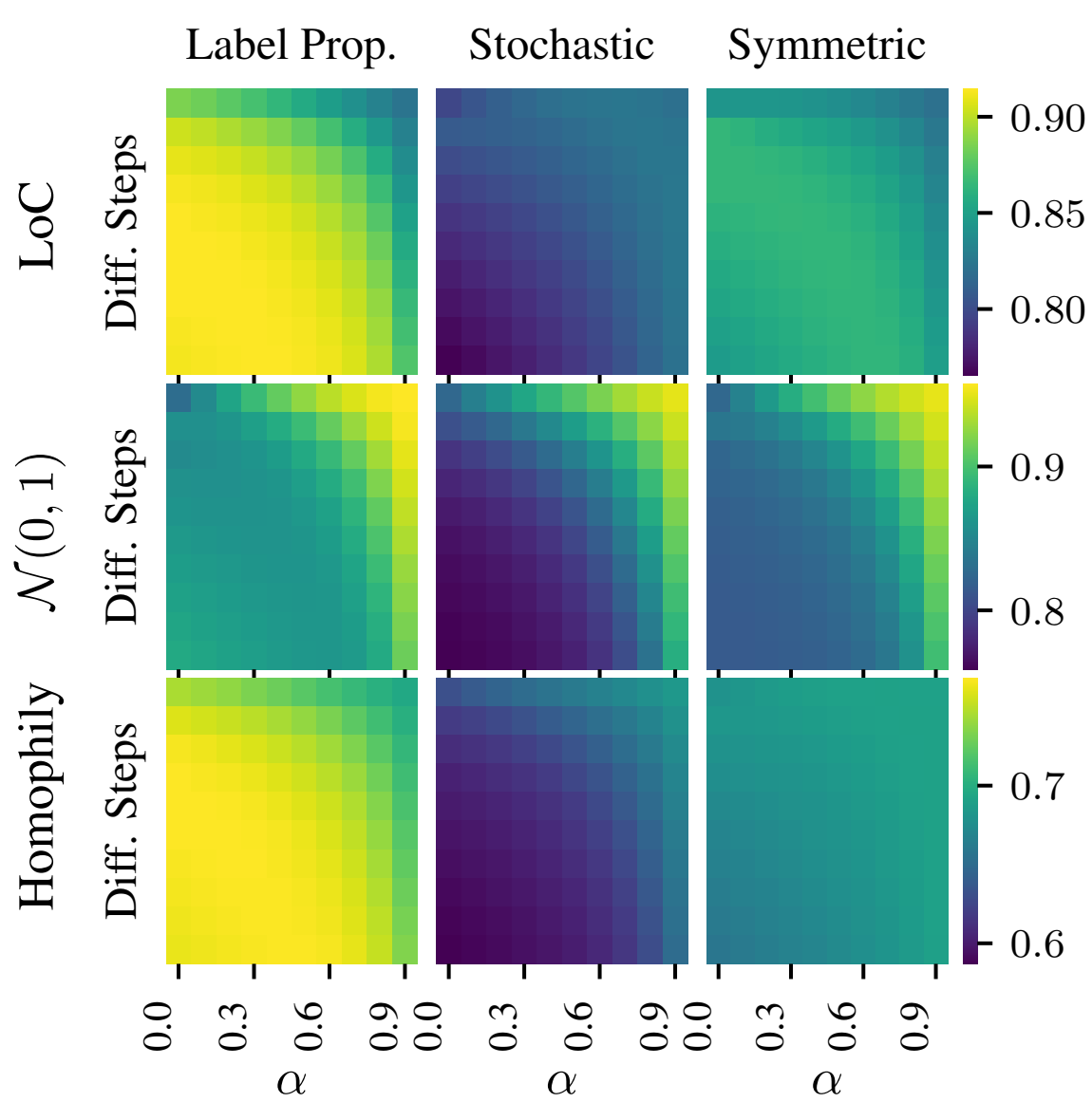

Figure 8: O.o.d. detection AUC of different diffusion operators on Leave-out-Classes (top), $\mathcal{N}(0,1)$ (middle) and homophily (bottom) shifts.

| Dataset | Model | LoC (last) Clean / I.d. / O.o.D. | LoC (heterophilic) Clean / I.d. / O.o.D. | Ber($\hat{p}$) (near) Clean / I.d. / O.o.D. | Ber(0.5) Clean / I.d. / O.o.D. | $\mathcal{N}(0,1)$ (far) Clean / I.d. / O.o.D. | Page Rank Clean / I.d. / O.o.D. | Homophily Clean / I.d. / O.o.D. |
|---|---|---|---|---|---|---|---|---|
| CoraML | BGCN | 6.6±1.0 / 6.7±1.1 / 75.5±3.7 | 9.5±2.4 / 9.2±2.2 / 75.1±3.6 | 9.7±1.5 / 8.3±4.0 / 8.9±3.3 | 9.7±1.5 / 11.7±5.5 / 20.1±12.0 | 9.7±1.5 / 53.0±4.9 / 61.9±3.9 | 11.5±1.7 / 11.4±1.4 / 8.9±1.6 | 3.6±0.6 / 3.6±0.7 / 16.6±3.0 |
| | GCN-MCD | 6.0±1.1 / 6.0±1.1 / 56.8±2.5 | 7.0±1.5 / 7.0±1.5 / 54.9±3.5 | 8.4±1.3 / 20.3±2.4 / 22.3±3.4 | 8.4±1.3 / 20.0±2.9 / 16.8±4.1 | 8.4±1.3 / 48.7±3.6 / 67.1±1.3 | 7.3±2.0 / 7.3±2.0 / 14.8±2.4 | 9.3±1.4 / 9.3±1.4 / 7.0±1.1 |
| | GCN-DE | 11.7±2.1 / 11.5±2.7 / 50.7±1.6 | 10.6±1.7 / 10.6±2.0 / 49.2±2.1 | 15.1±2.5 / 24.3±3.3 / 17.9±2.9 | 15.1±2.5 / 21.8±4.1 / 14.6±4.4 | 15.1±2.5 / 38.3±3.0 / 59.5±2.1 | 12.1±1.9 / 12.6±1.9 / 19.3±2.3 | 19.2±2.0 / 19.7±2.3 / 11.6±2.3 |
| | GCN-Ens | 5.7±1.0 / 5.7±1.0 / 56.8±2.5 | 7.1±1.6 / 7.1±1.6 / 54.9±3.8 | 9.0±0.7 / 21.1±2.4 / 23.0±2.6 | 9.0±0.7 / 21.2±2.6 / 17.9±3.3 | 9.0±0.7 / 45.5±6.0 / 63.8±3.1 | 7.4±1.3 / 7.4±1.3 / 14.7±1.9 | 9.4±1.4 / 9.4±1.4 / 7.1±1.0 |
| | GCN-MCD+DE | 11.0±2.0 / 11.0±1.9 / 51.6±2.2 | 9.7±1.8 / 10.2±1.9 / 50.2±2.4 | 15.6±2.9 / 23.7±3.0 / 18.3±4.1 | 15.6±2.9 / 21.0±4.5 / 13.8±4.7 | 15.6±2.9 / 36.7±2.8 / 57.3±1.8 | 11.4±2.0 / 11.9±2.0 / 17.6±2.4 | 19.4±2.2 / 18.7±2.0 / 12.0±2.6 |
| | SGCN | 28.5±3.3 / 28.5±3.3 / 38.5±1.7 | 26.9±3.8 / 26.9±3.8 / 39.5±1.6 | 27.4±18.3 / 29.9±23.5 / 24.6±20.5 | 27.4±18.3 / 27.7±24.2 / 21.0±16.9 | 27.4±18.3 / 11.3±8.0 / 23.0±20.9 | 40.5±1.4 / 40.5±1.4 / 49.7±1.9 | 49.6±2.1 / 49.6±2.1 / 38.0±2.8 |
| | GPN | 7.6±1.4 / 7.6±1.4 / 56.5±1.9 | 10.3±2.3 / 10.3±2.3 / 54.4±4.0 | 12.2±1.9 / 14.1±2.9 / 13.0±3.3 | 12.2±1.9 / 15.1±3.6 / 13.2±3.9 | 12.2±1.9 / 12.4±1.9 / 10.6±1.5 | 11.3±1.3 / 11.3±1.3 / 10.9±2.2 | 10.0±1.1 / 10.0±1.1 / 8.6±1.5 |
| | GCN (Ours) | 5.9±1.2 / 5.9±1.2 / 56.8±2.8 | 6.9±1.6 / 6.9±1.6 / 54.9±3.5 | 8.5±1.4 / 20.4±2.7 / 22.3±2.7 | 8.5±1.4 / 20.5±3.0 / 18.3±4.3 | 8.5±1.4 / 49.8±2.8 / 68.6±3.2 | 7.6±1.4 / 7.6±1.4 / 14.5±2.9 | 9.6±1.5 / 9.6±1.5 / 7.5±1.0 |
| CoraML LLM | BGCN | 3.9±1.1 / 4.0±1.3 / 71.6±4.5 | 4.8±1.6 / 4.7±1.5 / 70.8±9.0 | n.a. / n.a. / n.a. | 6.4±1.3 / 14.8±3.7 / 24.0±4.0 | 6.4±1.3 / 21.6±3.2 / 43.7±3.3 | 6.0±1.2 / 6.1±1.2 / 7.6±1.9 | 3.2±0.7 / 3.3±0.7 / 15.3±2.3 |
| | GCN-MCD | 4.5±0.7 / 4.5±0.7 / 64.4±2.3 | 4.3±0.8 / 4.3±0.8 / 65.3±5.1 | n.a. / n.a. / n.a. | 6.4±1.7 / 17.5±2.1 / 25.3±2.4 | 6.4±1.7 / 20.2±2.7 / 45.7±3.1 | 5.8±0.9 / 5.8±0.9 / 10.6±1.7 | 5.0±0.7 / 5.0±0.7 / 10.8±1.6 |
| | GCN-DE | 4.9±1.1 / 5.2±0.9 / 61.4±2.3 | 5.5±1.5 / 5.3±1.3 / 62.0±4.8 | n.a. / n.a. / n.a. | 8.1±1.8 / 18.4±2.5 / 22.7±3.0 | 8.1±1.8 / 17.9±3.4 / 41.7±3.8 | 7.9±1.2 / 8.0±1.3 / 11.6±2.0 | 7.7±1.3 / 8.0±1.2 / 8.6±1.7 |
| | GCN-Ens | 4.2±0.6 / 4.2±0.6 / 64.4±2.3 | 4.4±0.8 / 4.4±0.8 / 65.3±5.1 | n.a. / n.a. / n.a. | 6.8±1.7 / 18.0±1.7 / 26.3±2.9 | 6.8±1.7 / 17.4±4.9 / 45.8±2.5 | 5.6±1.1 / 5.6±1.1 / 10.9±1.3 | 4.9±0.5 / 4.9±0.5 / 10.8±1.6 |
| | GCN-MCD+DE | 5.1±1.1 / 5.0±1.4 / 61.4±1.9 | 5.6±1.3 / 5.6±1.5 / 61.7±3.7 | n.a. / n.a. / n.a. | 8.4±1.4 / 18.0±2.9 / 22.3±2.8 | 8.4±1.4 / 17.8±3.3 / 42.9±2.7 | 7.7±1.2 / 7.2±1.2 / 10.9±1.6 | 7.8±1.3 / 8.0±1.4 / 8.7±2.1 |
| | SGCN | 21.1±1.2 / 21.1±1.2 / 41.1±1.0 | 24.5±2.0 / 24.5±2.0 / 41.7±3.0 | n.a. / n.a. / n.a. | 36.3±11.3 / 43.3±16.2 / 41.6±17.4 | 36.3±11.3 / 17.8±7.6 / 12.2±5.4 | 35.4±2.5 / 35.4±2.5 / 45.9±2.2 | 42.6±1.5 / 42.6±1.5 / 30.6±2.1 |
| | GPN | 6.4±1.0 / 6.4±1.0 / 56.8±1.0 | 8.1±1.4 / 8.1±1.4 / 55.7±5.2 | n.a. / n.a. / n.a. | 8.9±1.7 / 10.0±2.2 / 11.3±2.7 | 8.9±1.7 / 9.5±1.7 / 9.3±1.7 | 9.2±1.3 / 9.2±1.3 / 8.8±1.8 | 9.7±1.2 / 9.7±1.2 / 8.8±2.1 |
| | GCN (Ours) | 4.2±0.6 / 4.2±0.6 / 64.4±2.5 | 4.6±1.1 / 4.6±1.1 / 65.0±5.6 | n.a. / n.a. / n.a. | 6.7±1.8 / 18.2±2.1 / 27.0±2.4 | 6.7±1.8 / 18.8±4.0 / 43.2±3.6 | 5.9±1.5 / 5.9±1.5 / 10.5±2.5 | 5.0±0.6 / 5.0±0.6 / 10.7±1.7 |
| Citeseer | BGCN | 8.4±1.3 / 8.4±1.2 / 66.4±2.1 | 4.0±0.6 / 3.9±0.7 / 71.1±3.2 | 8.6±1.0 / 8.5±1.5 / 12.5±2.4 | 8.6±1.0 / 13.3±3.8 / 29.0±11.8 | 8.6±1.0 / 37.8±3.5 / 53.2±2.5 | 8.0±0.9 / 7.9±1.0 / 5.4±1.2 | 5.4±1.0 / 5.4±1.1 / 8.5±1.5 |
| | GCN-MCD | 3.8±0.6 / 3.8±0.6 / 54.4±1.5 | 5.2±1.4 / 5.2±1.4 / 58.6±3.0 | 5.6±0.8 / 9.2±1.3 / 8.1±1.7 | 5.6±0.8 / 8.4±1.7 / 10.4±4.7 | 5.6±0.8 / 40.5±2.9 / 68.9±2.5 | 5.3±1.1 / 5.3±1.1 / 12.3±2.2 | 5.1±1.1 / 5.1±1.1 / 5.7±0.7 |
| | GCN-DE | 6.9±1.2 / 7.1±1.6 / 50.8±2.6 | 9.2±1.9 / 8.9±1.4 / 53.7±2.2 | 9.3±1.4 / 10.8±2.0 / 8.0±1.8 | 9.3±1.4 / 9.8±2.4 / 14.0±7.5 | 9.3±1.4 / 28.4±3.0 / 58.3±2.2 | 10.3±1.8 / 10.1±1.7 / 16.2±2.5 | 10.0±1.3 / 10.1±1.5 / 7.7±1.4 |
| | GCN-Ens | 3.9±0.6 / 3.9±0.6 / 53.9±1.4 | 5.2±1.4 / 5.2±1.4 / 58.4±2.8 | 6.0±0.8 / 9.0±1.0 / 8.8±1.8 | 6.0±0.8 / 9.4±2.3 / 11.5±2.5 | 6.0±0.8 / 38.3±2.0 / 63.3±1.6 | 5.0±0.8 / 5.0±0.8 / 12.0±2.1 | 5.0±1.1 / 5.0±1.1 / 5.7±0.7 |
| | GCN-MCD+DE | 6.8±1.3 / 6.8±1.3 / 50.9±2.1 | 8.6±1.8 / 8.7±1.4 / 54.2±2.1 | 8.8±1.5 / 10.3±2.1 / 8.1±1.8 | 8.8±1.5 / 9.3±2.7 / 12.8±5.1 | 8.8±1.5 / 29.2±2.6 / 59.6±1.8 | 9.9±1.9 / 10.0±2.5 / 15.7±2.8 | 10.3±1.7 / 10.2±2.0 / 8.2±1.7 |
| | SGCN | 25.0±2.0 / 25.0±2.0 / 36.1±1.7 | 29.6±1.6 / 29.6±1.6 / 37.0±1.1 | 38.0±1.9 / 40.8±2.0 / 30.2±2.4 | 38.0±1.9 / 40.8±3.4 / 29.1±7.9 | 38.0±1.9 / 23.2±2.2 / 52.0±2.3 | 35.8±1.6 / 35.8±1.6 / 47.9±3.0 | 39.2±2.2 / 39.2±2.2 / 37.2±1.4 |
| | GPN | 4.8±1.2 / 4.8±1.2 / 61.4±3.0 | 3.2±0.6 / 3.2±0.6 / 66.7±3.7 | 4.8±1.0 / 7.0±1.8 / 7.8±1.8 | 4.8±1.0 / 6.4±1.8 / 10.5±4.0 | 4.8±1.0 / 5.4±0.7 / 6.0±1.1 | 5.1±1.0 / 5.1±1.0 / 4.8±1.4 | 4.7±0.9 / 4.7±0.9 / 5.1±1.3 |
| | GCN (Ours) | 4.1±0.8 / 4.1±0.8 / 54.2±1.5 | 5.2±1.1 / 5.2±1.1 / 58.8±2.8 | 5.5±1.0 / 9.2±1.3 / 8.4±1.9 | 5.5±1.0 / 8.2±1.8 / 10.8±5.3 | 5.5±1.0 / 41.8±1.7 / 67.7±2.2 | 5.3±1.0 / 5.3±1.0 / 12.1±2.2 | 5.1±1.0 / 5.1±1.0 / 5.4±0.8 |
| PubMed | BGCN | 5.7±1.1 / 5.7±1.2 / 92.5±1.5 | 10.2±1.6 / 10.2±1.6 / 90.2±2.3 | n.a. / n.a. / n.a. | 10.2±4.4 / 10.0±3.8 / 16.7±9.2 | 10.2±4.4 / 37.9±6.3 / 52.5±2.6 | 12.4±5.4 / 12.4±5.4 / 8.4±5.6 | 5.6±1.7 / 5.7±1.7 / 21.3±2.2 |
| | GCN-MCD | 2.4±0.6 / 2.4±0.6 / 83.3±1.6 | 2.2±0.5 / 2.2±0.5 / 77.2±3.0 | n.a. / n.a. / n.a. | 3.8±1.0 / 9.2±2.5 / 8.7±2.8 | 3.8±1.0 / 28.5±2.0 / 49.0±1.2 | 3.3±0.6 / 3.3±0.6 / 8.6±1.3 | 2.7±0.9 / 2.7±0.9 / 14.2±1.7 |
| | GCN-DE | 3.2±1.0 / 3.1±0.8 / 82.0±1.5 | 2.3±0.7 / 2.4±0.8 / 73.3±2.3 | n.a. / n.a. / n.a. | 4.5±1.2 / 7.7±2.5 / 6.1±4.0 | 4.5±1.2 / 24.8±1.7 / 50.2±1.8 | 4.3±1.9 / 4.2±1.6 / 7.9±1.8 | 4.6±1.8 / 4.5±1.6 / 9.6±2.1 |
| | GCN-Ens | 2.3±0.5 / 2.3±0.5 / 83.3±1.5 | 2.2±0.6 / 2.2±0.6 / 77.4±2.6 | n.a. / n.a. / n.a. | 3.7±0.6 / 8.9±2.3 / 8.4±1.9 | 3.7±0.6 / 29.6±1.6 / 49.4±1.1 | 3.1±0.7 / 3.1±0.7 / 9.3±1.3 | 2.6±1.0 / 2.6±1.0 / 14.2±1.7 |
| | GCN-MCD+DE | 3.4±1.1 / 3.5±1.1 / 80.8±2.1 | 2.3±0.6 / 2.1±0.5 / 73.3±2.5 | n.a. / n.a. / n.a. | 5.0±1.7 / 8.3±2.5 / 5.3±3.3 | 5.0±1.7 / 24.0±2.4 / 49.4±1.4 | 3.9±1.7 / 4.1±1.8 / 7.5±2.0 | 4.3±1.3 / 4.3±1.3 / 9.9±1.4 |
| | SGCN | 10.3±1.1 / 10.3±1.1 / 74.9±1.0 | 8.4±1.6 / 8.4±1.6 / 69.6±1.5 | n.a. / n.a. / n.a. | 18.1±1.9 / 24.4±1.6 / 18.6±3.7 | 18.1±1.9 / 14.8±1.8 / 39.6±2.1 | 14.6±2.7 / 14.6±2.7 / 24.4±2.5 | 22.1±1.5 / 22.1±1.5 / 6.3±1.7 |
| | GPN | 4.5±0.9 / 4.5±0.9 / 80.8±1.1 | 3.0±1.2 / 3.0±1.2 / 78.2±3.2 | n.a. / n.a. / n.a. | 4.1±1.7 / 6.9±2.8 / 8.7±3.4 | 4.1±1.7 / 4.8±2.1 / 4.9±2.5 | 4.5±2.4 / 4.5±2.4 / 3.8±2.1 | 4.8±1.4 / 4.8±1.4 / 11.2±1.9 |
| | GCN (Ours) | 2.3±0.6 / 2.3±0.6 / 83.3±1.6 | 2.3±0.6 / 2.3±0.6 / 77.5±2.7 | n.a. / n.a. / n.a. | 3.9±0.8 / 8.8±2.7 / 8.4±3.1 | 3.9±0.8 / 29.4±2.6 / 50.2±1.8 | 3.3±0.8 / 3.3±0.8 / 9.4±1.5 | 2.8±1.1 / 2.8±1.1 / 14.3±2.2 |
| Amazon Photo | BGCN | 3.5±0.6 / 3.5±0.6 / 84.1±5.5 | 1.9±0.2 / 1.9±0.3 / 84.9±7.0 | 4.2±1.0 / 5.7±1.6 / 6.1±1.9 | 4.2±1.0 / 23.6±14.4 / 31.0±18.9 | 4.2±1.0 / 27.6±3.8 / 37.6±2.9 | 3.3±0.8 / 3.5±0.9 / 5.8±1.5 | 1.3±0.4 / 1.3±0.4 / 10.8±2.6 |
| | GCN-MCD | 3.6±0.9 / 3.6±0.9 / 74.8±3.9 | 4.2±1.2 / 4.2±1.2 / 72.2±8.5 | 5.7±1.3 / 21.7±2.4 / 22.7±2.4 | 5.7±1.3 / 9.8±3.1 / 15.8±9.1 | 5.7±1.3 / 21.1±7.3 / 33.0±5.7 | 4.0±0.9 / 4.0±0.9 / 17.6±1.6 | 5.6±0.6 / 5.6±0.6 / 5.0±1.0 |
| | GCN-DE | 5.1±1.0 / 4.8±1.1 / 71.5±3.5 | 6.4±1.1 / 6.2±1.1 / 69.1±7.4 | 8.6±1.9 / 25.5±2.9 / 25.4±2.9 | 8.6±1.9 / 8.9±3.8 / 11.4±5.2 | 8.6±1.9 / 21.8±4.7 / 33.7±4.5 | 6.4±1.1 / 6.4±0.9 / 18.9±2.1 | 9.0±0.6 / 8.9±0.7 / 5.1±0.8 |
| | GCN-Ens | 3.6±0.7 / 3.6±0.7 / 74.9±3.5 | 4.4±1.2 / 4.4±1.2 / 71.8±8.5 | 5.9±1.1 / 22.0±2.2 / 23.1±2.0 | 5.9±1.1 / 9.3±2.4 / 16.0±8.5 | 5.9±1.1 / 20.2±2.7 / 33.1±3.1 | 4.3±0.9 / 4.3±0.9 / 17.7±1.9 | 5.6±0.4 / 5.6±0.4 / 4.9±0.8 |
| | GCN-MCD+DE | 4.9±0.9 / 4.9±1.1 / 72.4±3.8 | 6.3±1.3 / 6.4±1.4 / 68.9±7.8 | 8.3±1.3 / 24.9±2.4 / 24.9±2.3 | 8.3±1.3 / 8.9±3.5 / 11.8±5.6 | 8.3±1.3 / 24.3±5.9 / 34.9±5.7 | 5.9±1.3 / 6.0±1.2 / 18.5±2.3 | 9.3±1.2 / 9.3±1.2 / 5.3±0.9 |
| | SGCN | 29.5±1.7 / 29.5±1.7 / 44.4±3.2 | 32.1±1.2 / 32.1±1.2 / 45.1±4.1 | 6.1±4.7 / 6.3±4.4 / 6.0±4.2 | 6.1±4.7 / 5.5±3.8 / 5.3±3.8 | 6.1±4.7 / 5.6±3.5 / 4.9±3.2 | 28.6±16.6 / 28.6±16.6 / 38.7±24.2 | 5.5±4.4 / 5.5±4.4 / 7.0±5.4 |
| | GPN | 12.3±1.7 / 12.3±1.7 / 58.1±3.8 | 5.6±1.2 / 5.6±1.2 / 65.6±3.9 | 18.1±2.9 / 19.4±3.9 / 19.6±5.0 | 18.1±2.9 / 19.6±3.8 / 19.7±4.6 | 18.1±2.9 / 17.9±3.3 / 18.0±3.9 | 18.6±2.6 / 18.6±2.6 / 15.8±2.1 | 18.3±1.4 / 18.3±1.4 / 16.4±2.8 |
| | GCN (Ours) | 3.3±0.8 / 3.3±0.8 / 75.4±3.3 | 4.4±1.1 / 4.4±1.1 / 71.7±8.7 | 5.7±1.0 / 21.8±2.0 / 23.0±2.2 | 5.7±1.0 / 9.2±2.9 / 15.3±8.8 | 5.7±1.0 / 21.3±4.8 / 33.4±4.2 | 4.1±1.0 / 4.1±1.0 / 17.7±2.0 | 5.5±0.6 / 5.5±0.6 / 5.1±1.0 |
| Amazon Computers | BGCN | 5.3±1.8 / 5.3±1.8 / 76.3±9.5 | 3.2±1.1 / 3.2±1.1 / 75.5±8.0 | 7.0±2.2 / 9.5±3.6 / 10.2±3.8 | 7.0±2.2 / 21.1±8.6 / 27.4±12.3 | 7.0±2.2 / 24.9±3.2 / 36.4±2.7 | 5.4±1.7 / 5.2±1.4 / 12.5±2.7 | 3.0±0.7 / 3.0±0.6 / 15.4±2.9 |
| | GCN-MCD | 8.6±3.2 / 8.6±3.2 / 61.5±7.6 | 5.8±1.2 / 5.8±1.2 / 60.6±6.0 | 7.9±1.4 / 27.2±1.6 / 28.8±1.6 | 7.9±1.4 / 15.9±3.6 / 16.1±3.2 | 7.9±1.4 / 18.0±2.8 / 30.8±3.7 | 5.8±1.5 / 5.8±1.5 / 26.1±2.5 | 8.0±1.1 / 8.0±1.1 / 7.7±0.8 |
| | GCN-DE | 10.0±3.4 / 10.0±3.5 / 58.8±7.0 | 7.5±1.7 / 7.5±1.6 / 57.8±5.6 | 10.6±2.1 / 29.9±1.9 / 29.9±1.9 | 10.6±2.1 / 17.9±5.2 / 15.4±4.3 | 10.6±2.1 / 19.9±3.2 / 31.1±3.2 | 7.9±1.7 / 8.0±1.8 / 25.4±2.4 | 10.9±1.6 / 10.9±1.6 / 6.8±1.2 |
| | GCN-Ens | 8.2±2.8 / 8.2±2.8 / 61.7±7.9 | 5.5±1.0 / 5.5±1.0 / 61.3±6.7 | 7.7±1.3 / 27.1±1.3 / 29.0±1.9 | 7.7±1.3 / 16.3±3.1 / 15.9±2.4 | 7.7±1.3 / 19.9±3.6 / 29.5±4.0 | 6.0±1.1 / 6.0±1.1 / 26.6±1.7 | 8.0±0.7 / 8.0±0.7 / 7.6±0.9 |
| | GCN-MCD+DE | 10.4±3.4 / 10.5±3.4 / 57.6±7.0 | 7.5±1.7 / 7.6±1.6 / 57.1±6.3 | 10.1±2.0 / 29.0±2.2 / 29.4±1.9 | 10.1±2.0 / 16.8±4.8 / 14.2±4.1 | 10.1±2.0 / 18.7±3.2 / 30.2±4.0 | 7.8±1.4 / 8.0±1.4 / 25.6±1.6 | 10.9±2.1 / 10.9±2.0 / 6.7±1.0 |
| | SGCN | 29.6±11.5 / 29.6±11.5 / 24.2±8.1 | 35.5±2.8 / 35.5±2.8 / 39.4±6.8 | 12.8±9.8 / 12.9±9.7 / 13.0±9.7 | 12.8±9.8 / 12.9±9.7 / 13.1±9.7 | 12.8±9.8 / 11.9±9.1 / 10.9±8.2 | 10.6±9.2 / 10.6±9.2 / 11.8±9.7 | 9.7±7.6 / 9.7±7.6 / 9.2±7.5 |
| | GPN | 19.3±4.7 / 19.3±4.7 / 42.0±3.6 | 11.6±1.9 / 11.6±1.9 / 60.2±4.4 | 20.1±3.7 / 22.8±3.0 / 24.5±3.1 | 20.1±3.7 / 22.3±2.9 / 23.9±3.0 | 20.1±3.7 / 20.7±2.9 / 22.2±3.2 | 17.9±2.9 / 17.9±2.9 / 17.7±2.8 | 17.9±3.6 / 17.9±3.6 / 11.4±3.0 |
| | GCN (Ours) | 8.2±2.7 / 8.2±2.7 / 61.8±7.5 | 5.5±1.2 / 5.5±1.2 / 61.4±6.2 | 7.7±1.5 / 26.7±1.8 / 28.7±1.7 | 7.7±1.5 / 16.4±3.5 / 16.1±2.6 | 7.7±1.5 / 18.1±3.0 / 30.1±3.3 | 6.2±1.5 / 6.2±1.5 / 26.9±2.1 | 7.7±0.9 / 7.7±0.9 / 7.9±0.9 |
| Coauthor Physics | BGCN | 1.0±0.4 / 1.0±0.5 / 80.1±2.8 | 0.7±0.3 / 0.8±0.3 / 83.7±5.3 | 2.2±0.7 / 3.9±1.3 / 5.2±1.9 | 2.2±0.7 / 13.1±7.1 / 18.4±12.1 | 2.2±0.7 / 17.4±9.8 / 20.8±8.8 | 1.6±0.4 / 1.6±0.5 / 1.5±0.4 | 1.1±0.2 / 1.2±0.4 / 3.5±0.7 |
| | GCN-MCD | 2.6±0.5 / 2.6±0.5 / 70.0±2.8 | 2.2±0.3 / 2.2±0.3 / 71.1±3.6 | 3.0±0.7 / 14.7±1.5 / 16.2±2.5 | 3.0±0.7 / 17.7±1.2 / 20.4±2.3 | 3.0±0.7 / 20.9±1.7 / 36.8±1.9 | 2.4±0.5 / 2.4±0.5 / 9.7±1.1 | 3.9±0.6 / 3.9±0.6 / 1.4±0.3 |
| | GCN-DE | 3.2±0.6 / 3.2±0.6 / 68.6±2.8 | 3.0±0.5 / 3.1±0.5 / 69.0±3.3 | 4.6±0.6 / 15.8±1.6 / 14.7±3.0 | 4.6±0.6 / 19.3±1.1 / 20.5±3.1 | 4.6±0.6 / 25.6±1.7 / 39.3±1.9 | 3.4±0.6 / 3.4±0.5 / 9.7±1.2 | 5.5±0.7 / 5.5±0.7 / 2.0±0.5 |
| | GCN-Ens | 2.6±0.5 / 2.6±0.5 / 70.1±2.7 | 2.2±0.2 / 2.2±0.2 / 71.0±3.7 | 3.1±0.4 / 14.7±0.8 / 16.6±2.2 | 3.1±0.4 / 17.8±0.7 / 20.9±2.2 | 3.1±0.4 / 20.4±1.6 / 35.2±1.4 | 2.4±0.3 / 2.4±0.3 / 9.6±0.9 | 3.9±0.5 / 3.9±0.5 / 1.4±0.3 |
| | GCN-MCD+DE | 3.5±0.9 / 3.5±0.9 / 68.6±3.2 | 2.9±0.4 / 2.8±0.5 / 69.1±3.1 | 4.8±0.5 / 16.3±1.3 / 15.4±2.8 | 4.8±0.5 / 19.7±1.2 / 20.0±3.7 | 4.8±0.5 / 25.5±1.9 / 39.9±2.3 | 3.3±0.6 / 3.3±0.6 / 9.6±1.3 | 5.5±0.8 / 5.5±0.7 / 2.1±0.6 |
| | SGCN | 23.1±1.3 / 23.1±1.3 / 48.8±1.7 | 23.1±1.1 / 23.1±1.1 / 47.3±1.9 | 38.3±1.5 / 53.5±1.1 / 52.5±2.1 | 38.3±1.5 / 53.5±2.7 / 51.2±5.2 | 38.3±1.5 / 15.2±2.2 / 14.6±1.3 | 32.7±1.3 / 32.7±1.3 / 46.5±1.2 | 39.0±2.1 / 39.0±2.1 / 34.4±1.8 |
| | GPN | 9.2±0.8 / 9.2±0.8 / 65.2±2.2 | 5.5±0.7 / 5.5±0.7 / 77.7±3.0 | 11.5±0.9 / 13.2±1.6 / 15.5±2.2 | 11.5±0.9 / 11.3±1.2 / 12.7±1.6 | 11.5±0.9 / 11.1±0.8 / 12.3±1.0 | 12.3±1.2 / 12.3±1.2 / 10.5±1.1 | 11.2±0.9 / 11.2±0.9 / 10.3±1.3 |
| | GCN (Ours) | 2.6±0.6 / 2.6±0.6 / 70.0±2.9 | 2.2±0.2 / 2.2±0.2 / 71.0±3.5 | 2.9±0.5 / 14.6±1.2 / 16.2±2.5 | 2.9±0.5 / 17.6±1.1 / 20.6±2.6 | 2.9±0.5 / 21.2±1.7 / 37.1±2.0 | 2.4±0.5 / 2.4±0.5 / 9.6±1.1 | 3.9±0.6 / 3.9±0.6 / 1.3±0.2 |
| Coauthor-CS | BGCN | 2.2±0.4 / 2.1±0.4 / 69.8±5.0 | 1.3±0.3 / 1.3±0.3 / 72.9±3.1 | 2.6±0.5 / 7.0±1.0 / 12.3±2.1 | 2.6±0.5 / 15.6±6.8 / 19.7±13.0 | 2.6±0.5 / 22.2±6.8 / 29.2±7.4 | 2.5±0.4 / 2.4±0.4 / 1.9±0.5 | 1.5±0.3 / 1.5±0.3 / 4.7±0.7 |
| | GCN-MCD | 5.5±1.4 / 5.5±1.4 / 53.0±4.1 | 4.3±0.5 / 4.3±0.5 / 57.2±3.2 | 5.4±0.6 / 23.9±1.1 / 32.6±1.2 | 5.4±0.6 / 24.1±2.0 / 29.1±5.1 | 5.4±0.6 / 35.2±3.1 / 57.8±2.5 | 4.6±0.8 / 4.6±0.8 / 14.5±1.7 | 6.2±0.5 / 6.2±0.5 / 3.9±0.5 |
| | GCN-DE | 9.4±1.3 / 9.5±1.5 / 48.7±3.2 | 7.9±0.8 / 7.7±0.8 / 52.6±2.7 | 10.3±0.9 / 29.6±1.5 / 33.2±1.6 | 10.3±0.9 / 29.6±2.5 / 26.2±4.4 | 10.3±0.9 / 28.1±1.6 / 46.4±1.3 | 7.5±0.8 / 7.5±0.9 / 17.3±1.4 | 10.8±0.9 / 10.8±0.8 / 7.5±1.0 |
| | GCN-Ens | 5.4±1.0 / 5.4±1.0 / 53.1±3.7 | 4.3±0.3 / 4.3±0.3 / 57.1±2.9 | 5.5±0.4 / 24.3±0.4 / 33.1±0.9 | 5.5±0.4 / 24.8±1.7 / 30.5±5.7 | 5.5±0.4 / 27.2±1.7 / 48.5±1.6 | 4.6±0.4 / 4.6±0.4 / 14.6±1.0 | 6.2±0.2 / 6.2±0.2 / 3.8±0.3 |
| | GCN-MCD+DE | 9.4±1.4 / 9.4±1.2 / 48.6±3.2 | 7.7±0.7 / 7.6±0.8 / 53.0±2.9 | 10.1±0.9 / 29.3±1.1 / 32.9±1.5 | 10.1±0.9 / 29.6±2.1 / 27.7±4.9 | 10.1±0.9 / 27.1±2.1 / 45.3±1.7 | 7.5±1.0 / 7.8±1.3 / 17.2±1.5 | 10.8±0.7 / 10.8±0.6 / 7.5±1.0 |
| | SGCN | 18.9±14.3 / 18.9±14.3 / 10.4±1.0 | 20.3±13.2 / 20.3±13.2 / 10.1±0.0 | 8.6±7.0 / 6.4±5.6 / 5.5±5.1 | 8.6±7.0 / 4.1±4.0 / 3.9±3.0 | 8.6±7.0 / 3.2±2.3 / 2.4±1.5 | 9.1±8.4 / 9.1±8.4 / 9.2±8.7 | 7.6±8.3 / 7.6±8.3 / 5.6±5.7 |
| | GPN | 19.4±1.8 / 19.4±1.8 / 49.5±3.2 | 20.1±1.5 / 20.1±1.5 / 44.6±2.7 | 20.7±1.9 / 22.6±2.3 / 23.8±2.7 | 20.7±1.9 / 20.6±1.9 / 20.4±2.1 | 20.7±1.9 / 20.5±1.8 / 20.1±2.0 | 22.0±1.4 / 22.0±1.4 / 19.1±1.4 | 21.1±1.3 / 21.1±1.3 / 18.2±2.1 |
| | GCN (Ours) | 5.2±1.0 / 5.2±1.0 / 53.4±3.6 | 4.3±0.4 / 4.3±0.4 / 57.2±2.9 | 5.5±0.6 / 24.1±1.0 / 32.5±1.3 | 5.5±0.6 / 24.2±2.4 / 27.7±5.7 | 5.5±0.6 / 34.7±3.1 / 57.2±2.5 | 4.5±0.5 / 4.5±0.5 / 14.6±1.2 | 6.2±0.4 / 6.2±0.4 / 3.7±0.4 |

Table 21: ECE (↓) for different backbones on clean data as well as i.d. and o.o.d. nodes after a distribution shift in a transductive setting (best and runner-up).

| | Model | LoC *(last)* Clean / I.d. / O.o.D. | LoC *(heterophilic)* Clean / I.d. / O.o.D. | Ber($\hat{p}$) *(near)* Clean / I.d. / O.o.D. | Ber(0.5) Clean / I.d. / O.o.D. | $\mathcal{N}(0,1)$ *(far)* Clean / I.d. / O.o.D. | Page Rank Clean / I.d. / O.o.D. | Homophily Clean / I.d. / O.o.D. |
|---|---|---|---|---|---|---|---|---|
| CoraML | BGCN | 0.17±0.0 / 0.16±0.0 / 1.59±0.0 | 0.23±0.0 / 0.23±0.0 / 1.57±0.1 | 0.40±0.0 / **0.39**±0.0 / **0.47**±0.0 | 0.40±0.0 / 0.56±0.1 / 0.73±0.1 | 0.40±0.0 / 1.18±0.1 / 1.30±0.0 | 0.32±0.0 / 0.31±0.0 / 0.32±0.0 | **0.17**±0.0 / **0.11**±0.0 / **0.42**±0.0 |
| | GCN-MCD | **0.15**±0.0 / **0.15**±0.0 / 1.40±0.0 | **0.20**±0.0 / **0.21**±0.0 / 1.38±0.0 | 0.35±0.0 / 0.43±0.0 / 0.53±0.0 | 0.35±0.0 / 0.44±0.0 / 0.55±0.0 | 0.35±0.0 / 1.09±0.0 / 1.41±0.0 | **0.26**±0.0 / **0.25**±0.0 / **0.31**±0.0 | 0.17±0.0 / **0.17**±0.0 / **0.45**±0.0 |
| | GCN-DE | 0.21±0.0 / 0.21±0.0 / **1.36**±0.0 | 0.28±0.0 / 0.29±0.0 / 1.35±0.0 | 0.44±0.0 / 0.53±0.0 / 0.66±0.0 | 0.44±0.0 / 0.56±0.0 / 0.71±0.0 | 0.44±0.0 / 1.02±0.0 / 1.31±0.0 | 0.33±0.0 / 0.32±0.0 / 0.40±0.0 | 0.29±0.0 / 0.29±0.0 / 0.55±0.0 |
| | GCN-Ens | **0.15**±0.0 / **0.15**±0.0 / 1.39±0.0 | **0.20**±0.0 / **0.20**±0.0 / 1.38±0.0 | **0.35**±0.0 / 0.42±0.0 / 0.53±0.0 | **0.35**±0.0 / **0.43**±0.0 / 0.55±0.0 | **0.35**±0.0 / 1.08±0.0 / 1.34±0.0 | **0.26**±0.0 / **0.25**±0.0 / **0.31**±0.0 | **0.17**±0.0 / **0.17**±0.0 / 0.45±0.0 |
| | GCN-MCD+DE | 0.21±0.0 / 0.21±0.0 / **1.36**±0.0 | 0.28±0.0 / 0.29±0.0 / **1.35**±0.0 | 0.44±0.0 / 0.53±0.0 / 0.66±0.0 | 0.44±0.0 / 0.56±0.0 / 0.71±0.0 | 0.44±0.0 / 1.07±0.0 / 1.30±0.0 | 0.33±0.0 / 0.32±0.0 / 0.40±0.0 | 0.29±0.0 / 0.29±0.0 / 0.54±0.0 |
| | SGCN | 0.31±0.0 / 0.33±0.0 / **1.27**±0.0 | 0.30±0.0 / 0.32±0.0 / **1.27**±0.0 | 0.80±0.1 / 0.82±0.1 / 0.84±0.0 | 0.80±0.1 / 0.83±0.1 / 0.84±0.0 | 0.80±0.1 / **0.86**±0.0 / **0.91**±0.1 | 0.55±0.1 / 0.53±0.1 / 0.65±0.1 | 0.53±0.0 / 0.58±0.0 / 0.72±0.0 |
| | GPN | 0.17±0.0 / 0.17±0.0 / 1.42±0.0 | 0.22±0.0 / 0.22±0.0 / 1.40±0.0 | 0.40±0.0 / **0.37**±0.0 / **0.38**±0.0 | 0.40±0.0 / **0.38**±0.0 / **0.40**±0.1 | 0.40±0.0 / **0.36**±0.0 / **0.35**±0.0 | 0.33±0.0 / 0.32±0.0 / 0.33±0.0 | **0.17**±0.0 / 0.26±0.1 / 0.63±0.1 |
| | GCN (Ours) | **0.15**±0.0 / **0.15**±0.0 / 1.40±0.0 | **0.20**±0.0 / **0.21**±0.0 / 1.39±0.0 | **0.35**±0.0 / 0.42±0.0 / 0.53±0.0 | **0.35**±0.0 / 0.43±0.0 / **0.54**±0.0 | **0.35**±0.0 / 1.12±0.0 / 1.40±0.0 | **0.26**±0.0 / 0.25±0.0 / 0.31±0.0 | 0.17±0.0 / 0.17±0.0 / 0.45±0.0 |
| CoraML LLM | BGCN | **0.15**±0.0 / **0.15**±0.0 / 1.59±0.0 | 0.16±0.0 / 0.16±0.0 / 1.54±0.1 | n.a. / n.a. / n.a. | 0.32±0.0 / **0.34**±0.0 / **0.45**±0.0 | 0.32±0.0 / 0.72±0.0 / 1.06±0.0 | 0.23±0.0 / 0.23±0.0 / **0.26**±0.0 | 0.14±0.0 / **0.08**±0.0 / 0.43±0.0 |
| | GCN-MCD | **0.14**±0.0 / **0.14**±0.0 / 1.51±0.0 | **0.15**±0.0 / **0.16**±0.0 / 1.49±0.1 | n.a. / n.a. / n.a. | 0.32±0.0 / 0.35±0.0 / 0.47±0.0 | 0.32±0.0 / 0.74±0.0 / 1.10±0.1 | **0.23**±0.0 / **0.22**±0.0 / 0.27±0.0 | 0.14±0.0 / **0.09**±0.0 / **0.40**±0.0 |
| | GCN-DE | 0.17±0.0 / 0.18±0.0 / 1.49±0.0 | 0.18±0.0 / 0.18±0.0 / 1.46±0.0 | n.a. / n.a. / n.a. | 0.35±0.0 / 0.40±0.0 / 0.55±0.0 | 0.35±0.0 / 0.75±0.0 / 1.13±0.0 | 0.26±0.0 / 0.25±0.0 / 0.31±0.0 | 0.17±0.0 / 0.15±0.0 / 0.43±0.0 |
| | GCN-Ens | **0.14**±0.0 / **0.14**±0.0 / 1.51±0.0 | **0.15**±0.0 / **0.16**±0.0 / 1.48±0.1 | n.a. / n.a. / n.a. | **0.32**±0.0 / 0.35±0.0 / 0.47±0.0 | **0.32**±0.0 / **0.71**±0.0 / 1.04±0.1 | **0.23**±0.0 / **0.22**±0.0 / **0.27**±0.0 | **0.13**±0.0 / **0.09**±0.0 / **0.40**±0.0 |
| | GCN-MCD+DE | 0.17±0.0 / 0.17±0.0 / 1.49±0.0 | 0.18±0.0 / 0.19±0.0 / 1.46±0.1 | n.a. / n.a. / n.a. | 0.35±0.0 / 0.39±0.0 / 0.55±0.0 | 0.35±0.0 / 0.75±0.0 / 1.10±0.1 | 0.26±0.0 / 0.25±0.0 / 0.31±0.0 | 0.17±0.0 / 0.15±0.0 / 0.43±0.0 |
| | SGCN | 0.24±0.0 / 0.26±0.0 / **1.30**±0.0 | 0.27±0.0 / 0.28±0.0 / **1.30**±0.0 | n.a. / n.a. / n.a. | 0.60±0.1 / 0.71±0.1 / 0.76±0.0 | 0.60±0.1 / 0.78±0.0 / **0.87**±0.0 | 0.48±0.1 / 0.45±0.1 / 0.58±0.1 | 0.48±0.2 / 0.53±0.1 / 0.67±0.1 |
| | GPN | 0.15±0.0 / 0.16±0.0 / **1.47**±0.0 | 0.18±0.0 / 0.18±0.0 / **1.42**±0.0 | n.a. / n.a. / n.a. | 0.34±0.0 / **0.32**±0.0 / **0.35**±0.0 | 0.34±0.0 / **0.31**±0.0 / **0.34**±0.0 | 0.28±0.0 / 0.27±0.0 / 0.30±0.0 | **0.13**±0.0 / 0.21±0.1 / 0.61±0.1 |
| | GCN (Ours) | **0.14**±0.0 / **0.14**±0.0 / 1.51±0.0 | **0.15**±0.0 / 0.16±0.0 / 1.48±0.1 | n.a. / n.a. / n.a. | **0.32**±0.0 / 0.36±0.0 / 0.47±0.0 | **0.32**±0.0 / 0.76±0.0 / 1.09±0.0 | 0.23±0.0 / 0.22±0.0 / 0.27±0.0 | 0.14±0.0 / 0.10±0.0 / **0.40**±0.0 |
| Citeseer | BGCN | 0.23±0.0 / 0.23±0.0 / 1.54±0.0 | 0.15±0.0 / 0.15±0.0 / 1.55±0.0 | 0.36±0.0 / 0.41±0.0 / 0.61±0.0 | 0.36±0.0 / 0.53±0.1 / 0.87±0.1 | 0.36±0.0 / 0.98±0.0 / 1.32±0.1 | 0.33±0.0 / 0.30±0.0 / **0.24**±0.0 | 0.33±0.0 / **0.22**±0.0 / **0.29**±0.0 |
| | GCN-MCD | 0.22±0.0 / 0.22±0.0 / 1.43±0.0 | 0.15±0.0 / **0.15**±0.0 / 1.43±0.0 | 0.34±0.0 / 0.41±0.0 / 0.60±0.0 | 0.34±0.0 / 0.46±0.0 / 0.71±0.1 | 0.34±0.0 / 0.97±0.1 / 1.39±0.0 | 0.31±0.0 / 0.28±0.0 / 0.26±0.0 | 0.32±0.0 / 0.24±0.0 / 0.29±0.0 |
| | GCN-DE | 0.28±0.0 / 0.28±0.0 / 1.40±0.0 | 0.22±0.0 / 0.22±0.0 / 1.40±0.0 | 0.40±0.0 / 0.47±0.0 / 0.71±0.0 | 0.40±0.0 / 0.50±0.0 / 0.75±0.0 | 0.40±0.0 / 0.87±0.0 / 1.30±0.0 | 0.36±0.0 / 0.34±0.0 / 0.34±0.0 | 0.37±0.0 / 0.32±0.0 / 0.37±0.0 |
| | GCN-Ens | **0.22**±0.0 / **0.22**±0.0 / 1.42±0.0 | **0.15**±0.0 / **0.15**±0.0 / 1.43±0.0 | **0.34**±0.0 / 0.41±0.0 / **0.60**±0.0 | **0.34**±0.0 / **0.43**±0.0 / **0.65**±0.0 | **0.34**±0.0 / 0.95±0.0 / 1.32±0.0 | **0.31**±0.0 / **0.28**±0.0 / 0.25±0.0 | **0.32**±0.0 / 0.24±0.0 / 0.29±0.0 |
| | GCN-MCD+DE | 0.28±0.0 / 0.28±0.0 / **1.40**±0.0 | 0.21±0.0 / 0.21±0.0 / **1.40**±0.0 | 0.40±0.0 / 0.48±0.0 / 0.71±0.0 | 0.40±0.0 / 0.49±0.0 / 0.74±0.1 | 0.40±0.0 / **0.86**±0.0 / 1.30±0.0 | 0.36±0.0 / 0.34±0.0 / 0.34±0.0 | 0.36±0.0 / 0.32±0.0 / 0.36±0.0 |
| | SGCN | 0.33±0.0 / 0.33±0.0 / **1.28**±0.0 | 0.30±0.0 / 0.30±0.0 / **1.28**±0.0 | 0.53±0.0 / 0.65±0.0 / 0.74±0.0 | 0.53±0.0 / 0.65±0.0 / 0.74±0.0 | 0.53±0.0 / 0.89±0.0 / **1.17**±0.0 | 0.47±0.0 / 0.45±0.0 / 0.54±0.0 | 0.51±0.0 / 0.52±0.0 / 0.55±0.0 |
| | GPN | **0.21**±0.0 / **0.21**±0.0 / 1.46±0.0 | **0.13**±0.0 / **0.13**±0.0 / 1.50±0.0 | **0.33**±0.0 / **0.32**±0.0 / **0.43**±0.0 | **0.33**±0.0 / **0.33**±0.1 / **0.47**±0.2 | **0.33**±0.0 / **0.27**±0.0 / **0.32**±0.0 | **0.31**±0.0 / 0.28±0.0 / **0.20**±0.0 | **0.31**±0.0 / **0.20**±0.0 / **0.28**±0.0 |
| | GCN (Ours) | 0.22±0.0 / 0.22±0.0 / 1.42±0.0 | 0.15±0.0 / 0.15±0.0 / 1.43±0.0 | 0.34±0.0 / **0.41**±0.0 / 0.60±0.0 | 0.34±0.0 / 0.44±0.0 / 0.69±0.1 | 0.34±0.0 / 0.99±0.0 / 1.38±0.0 | **0.31**±0.0 / **0.28**±0.0 / 0.25±0.0 | 0.32±0.0 / 0.24±0.0 / 0.30±0.0 |
| PubMed | BGCN | 0.13±0.0 / 0.13±0.0 / 1.91±0.0 | 0.29±0.1 / 0.29±0.1 / 1.76±0.1 | n.a. / n.a. / n.a. | **0.33**±0.0 / 0.40±0.0 / 0.57±0.1 | **0.33**±0.0 / 0.76±0.0 / 1.06±0.0 | 0.36±0.1 / 0.37±0.1 / 0.34±0.1 | 0.27±0.0 / **0.19**±0.0 / 0.50±0.0 |
| | GCN-MCD | 0.12±0.0 / **0.12**±0.0 / 1.76±0.0 | 0.26±0.0 / **0.26**±0.0 / 1.63±0.0 | n.a. / n.a. / n.a. | **0.33**±0.0 / 0.39±0.0 / 0.52±0.0 | **0.33**±0.0 / 0.72±0.0 / 1.04±0.0 | 0.31±0.0 / 0.31±0.0 / 0.31±0.0 | **0.25**±0.0 / **0.20**±0.0 / **0.43**±0.0 |
| | GCN-DE | 0.12±0.0 / 0.12±0.0 / 1.76±0.0 | 0.29±0.0 / 0.29±0.0 / **1.62**±0.0 | n.a. / n.a. / n.a. | 0.36±0.0 / 0.41±0.0 / 0.59±0.0 | 0.36±0.0 / 0.71±0.0 / 1.07±0.0 | 0.33±0.0 / 0.33±0.0 / 0.34±0.0 | 0.28±0.0 / 0.25±0.0 / 0.43±0.0 |
| | GCN-Ens | **0.12**±0.0 / **0.12**±0.0 / 1.77±0.0 | 0.26±0.0 / **0.26**±0.0 / 1.63±0.0 | n.a. / n.a. / n.a. | **0.33**±0.0 / **0.38**±0.0 / **0.52**±0.0 | **0.33**±0.0 / 0.71±0.0 / 1.02±0.0 | **0.30**±0.0 / **0.31**±0.0 / **0.31**±0.0 | **0.25**±0.0 / **0.20**±0.0 / **0.43**±0.0 |
| | GCN-MCD+DE | 0.12±0.0 / 0.13±0.0 / 1.76±0.0 | 0.29±0.0 / 0.29±0.0 / 1.62±0.0 | n.a. / n.a. / n.a. | 0.36±0.0 / 0.41±0.0 / 0.60±0.1 | 0.36±0.0 / 0.70±0.0 / 1.06±0.0 | 0.33±0.0 / 0.33±0.0 / 0.34±0.0 | 0.28±0.0 / 0.25±0.0 / 0.44±0.0 |
| | SGCN | 0.13±0.0 / 0.14±0.0 / **1.66**±0.0 | **0.26**±0.0 / **0.26**±0.0 / **1.59**±0.0 | n.a. / n.a. / n.a. | 0.37±0.0 / 0.45±0.0 / 0.54±0.0 | 0.37±0.0 / **0.65**±0.0 / **0.91**±0.0 | 0.33±0.0 / 0.32±0.0 / 0.37±0.0 | 0.30±0.0 / 0.33±0.0 / 0.45±0.0 |
| | GPN | **0.11**±0.0 / **0.12**±0.0 / **1.75**±0.0 | **0.22**±0.0 / **0.22**±0.0 / 1.67±0.0 | n.a. / n.a. / n.a. | 0.33±0.0 / **0.36**±0.0 / **0.46**±0.0 | 0.33±0.0 / **0.31**±0.0 / **0.36**±0.0 | **0.29**±0.0 / **0.29**±0.0 / **0.28**±0.0 | **0.21**±0.0 / 0.22±0.1 / 0.52±0.1 |
| | GCN (Ours) | **0.12**±0.0 / **0.12**±0.0 / 1.77±0.0 | 0.26±0.0 / **0.26**±0.0 / 1.63±0.0 | n.a. / n.a. / n.a. | 0.33±0.0 / 0.39±0.0 / 0.52±0.0 | 0.33±0.0 / 0.72±0.0 / 1.03±0.0 | 0.31±0.0 / 0.31±0.0 / 0.31±0.0 | 0.25±0.0 / **0.20**±0.0 / 0.43±0.0 |
| Amazon Photo | BGCN | **0.13**±0.0 / **0.13**±0.0 / 1.81±0.0 | **0.07**±0.0 / **0.07**±0.0 / 1.75±0.1 | **0.17**±0.0 / **0.27**±0.1 / **0.34**±0.1 | **0.17**±0.0 / 0.65±0.2 / 0.76±0.2 | **0.17**±0.0 / 0.89±0.0 / 1.02±0.0 | 0.15±0.0 / 0.13±0.0 / **0.18**±0.0 | **0.04**±0.0 / **0.05**±0.0 / 0.27±0.0 |
| | GCN-MCD | **0.12**±0.0 / **0.13**±0.0 / 1.62±0.0 | 0.09±0.0 / 0.09±0.0 / 1.59±0.1 | 0.17±0.0 / 0.28±0.0 / 0.35±0.0 | 0.17±0.0 / 0.48±0.1 / 0.57±0.1 | 0.17±0.0 / 0.86±0.0 / 1.02±0.0 | **0.13**±0.0 / **0.12**±0.0 / 0.23±0.0 | 0.05±0.0 / 0.08±0.0 / **0.26**±0.0 |
| | GCN-DE | 0.13±0.0 / 0.14±0.0 / 1.57±0.0 | 0.10±0.0 / 0.10±0.0 / 1.55±0.1 | 0.19±0.0 / 0.32±0.0 / 0.40±0.0 | 0.19±0.0 / **0.46**±0.1 / **0.55**±0.1 | 0.19±0.0 / 0.85±0.0 / 1.00±0.1 | 0.15±0.0 / 0.13±0.0 / 0.25±0.0 | 0.07±0.0 / 0.10±0.0 / 0.29±0.0 |
| | GCN-Ens | **0.12**±0.0 / **0.13**±0.0 / 1.62±0.0 | **0.08**±0.0 / **0.09**±0.0 / 1.59±0.1 | **0.17**±0.0 / **0.28**±0.0 / 0.34±0.0 | **0.17**±0.0 / 0.47±0.1 / 0.56±0.1 | **0.17**±0.0 / 0.83±0.1 / 0.98±0.0 | **0.13**±0.0 / **0.12**±0.0 / **0.23**±0.0 | 0.05±0.0 / 0.08±0.0 / **0.26**±0.0 |
| | GCN-MCD+DE | 0.14±0.0 / 0.14±0.0 / 1.57±0.0 | 0.10±0.0 / 0.10±0.0 / 1.55±0.1 | 0.20±0.0 / 0.33±0.0 / 0.41±0.0 | 0.20±0.0 / 0.47±0.1 / 0.56±0.1 | 0.20±0.0 / 0.85±0.1 / 1.00±0.0 | 0.14±0.0 / 0.13±0.0 / 0.25±0.0 | 0.07±0.0 / 0.10±0.0 / 0.29±0.0 |
| | SGCN | 0.28±0.0 / 0.30±0.0 / **1.27**±0.0 | 0.25±0.0 / 0.27±0.0 / **1.28**±0.0 | 0.87±0.0 / 0.87±0.0 / 0.87±0.0 | 0.87±0.0 / 0.87±0.0 / 0.87±0.0 | 0.87±0.0 / 0.87±0.0 / **0.87**±0.0 | 0.88±0.0 / 0.88±0.0 / 0.88±0.0 | 0.87±0.0 / 0.87±0.0 / 0.88±0.0 |
| | GPN | 0.20±0.0 / 0.20±0.0 / **1.47**±0.0 | 0.10±0.0 / 0.11±0.0 / **1.52**±0.1 | 0.27±0.0 / 0.28±0.0 / **0.32**±0.0 | 0.27±0.0 / **0.28**±0.0 / **0.32**±0.0 | 0.27±0.0 / **0.27**±0.0 / **0.30**±0.0 | 0.26±0.0 / 0.25±0.0 / 0.28±0.0 | **0.04**±0.0 / 0.17±0.0 / 0.41±0.1 |
| | GCN (Ours) | **0.12**±0.0 / **0.13**±0.0 / 1.62±0.0 | **0.08**±0.0 / **0.09**±0.0 / 1.60±0.1 | **0.17**±0.0 / 0.28±0.0 / 0.34±0.0 | **0.17**±0.0 / 0.47±0.1 / 0.56±0.1 | **0.17**±0.0 / **0.81**±0.0 / 0.96±0.0 | 0.14±0.0 / **0.12**±0.0 / 0.23±0.0 | 0.05±0.0 / **0.08**±0.0 / **0.26**±0.0 |
| Amazon Computers | BGCN | 0.24±0.1 / 0.24±0.1 / 1.65±0.1 | **0.11**±0.0 / **0.11**±0.0 / 1.63±0.1 | 0.33±0.0 / **0.39**±0.1 / **0.43**±0.1 | 0.33±0.0 / 0.56±0.2 / 0.64±0.2 | 0.33±0.0 / 0.95±0.1 / 1.09±0.1 | 0.32±0.0 / 0.31±0.0 / **0.33**±0.0 | 0.14±0.0 / **0.16**±0.0 / 0.46±0.0 |
| | GCN-MCD | **0.20**±0.0 / **0.20**±0.0 / 1.51±0.1 | **0.11**±0.0 / **0.12**±0.0 / 1.44±0.1 | **0.31**±0.0 / 0.42±0.0 / 0.47±0.0 | **0.31**±0.0 / 0.54±0.1 / 0.60±0.1 | **0.31**±0.0 / 0.85±0.0 / 0.99±0.0 | 0.29±0.0 / **0.27**±0.0 / 0.36±0.0 | 0.14±0.0 / 0.18±0.0 / **0.42**±0.0 |
| | GCN-DE | 0.21±0.0 / 0.21±0.0 / 1.46±0.0 | 0.12±0.0 / 0.13±0.0 / **1.41**±0.1 | 0.34±0.0 / 0.47±0.0 / 0.52±0.0 | 0.34±0.0 / 0.55±0.1 / 0.62±0.1 | 0.34±0.0 / 0.90±0.0 / 1.01±0.0 | 0.30±0.0 / 0.28±0.0 / 0.38±0.0 | 0.17±0.0 / 0.20±0.0 / 0.43±0.0 |
| | GCN-Ens | **0.20**±0.0 / **0.20**±0.0 / 1.51±0.0 | **0.11**±0.0 / **0.12**±0.0 / 1.44±0.1 | **0.31**±0.0 / 0.42±0.0 / 0.47±0.0 | **0.31**±0.0 / **0.54**±0.1 / **0.60**±0.1 | **0.31**±0.0 / **0.81**±0.0 / 0.94±0.0 | **0.29**±0.0 / **0.27**±0.0 / **0.36**±0.0 | **0.14**±0.0 / **0.18**±0.0 / **0.42**±0.0 |
| | GCN-MCD+DE | 0.21±0.0 / 0.21±0.0 / 1.46±0.0 | 0.12±0.0 / 0.13±0.0 / 1.42±0.1 | 0.34±0.0 / 0.47±0.0 / 0.52±0.0 | 0.34±0.0 / 0.54±0.1 / 0.61±0.1 | 0.34±0.0 / 0.90±0.0 / 1.01±0.0 | 0.30±0.0 / 0.28±0.0 / 0.38±0.0 | 0.17±0.0 / 0.20±0.0 / 0.43±0.0 |
| | SGCN | 0.77±0.1 / 0.77±0.1 / **1.17**±0.0 | 0.69±0.2 / 0.70±0.2 / **1.18**±0.0 | 0.90±0.0 / 0.90±0.0 / 0.90±0.0 | 0.90±0.0 / 0.90±0.0 / 0.90±0.0 | 0.90±0.0 / 0.90±0.0 / **0.90**±0.0 | 0.90±0.0 / 0.90±0.0 / 0.90±0.0 | 0.90±0.0 / 0.90±0.0 / 0.90±0.0 |
| | GPN | 0.26±0.0 / 0.26±0.0 / **1.34**±0.0 | 0.11±0.0 / 0.12±0.0 / 1.48±0.1 | 0.41±0.1 / **0.42**±0.1 / **0.46**±0.1 | 0.41±0.1 / **0.42**±0.1 / **0.45**±0.1 | 0.41±0.1 / **0.41**±0.1 / **0.44**±0.1 | 0.37±0.0 / 0.36±0.0 / 0.37±0.0 | **0.11**±0.0 / 0.32±0.1 / 0.54±0.1 |
| | GCN (Ours) | 0.20±0.0 / 0.20±0.0 / 1.50±0.1 | 0.11±0.0 / 0.12±0.0 / 1.44±0.1 | **0.32**±0.0 / 0.43±0.0 / 0.47±0.0 | **0.32**±0.0 / 0.54±0.1 / 0.60±0.1 | **0.32**±0.0 / 0.86±0.0 / 1.00±0.0 | **0.29**±0.0 / **0.27**±0.0 / 0.36±0.0 | 0.14±0.0 / 0.18±0.0 / 0.42±0.0 |
| Coauthor Physics | BGCN | **0.05**±0.0 / **0.05**±0.0 / 1.69±0.0 | **0.03**±0.0 / **0.03**±0.0 / 1.71±0.1 | 0.13±0.0 / **0.13**±0.0 / **0.18**±0.0 | 0.13±0.0 / 0.28±0.1 / 0.41±0.1 | 0.13±0.0 / 0.85±0.1 / 0.89±0.1 | **0.10**±0.0 / **0.09**±0.0 / **0.12**±0.0 | **0.03**±0.0 / **0.02**±0.0 / 0.16±0.0 |
| | GCN-MCD | **0.05**±0.0 / **0.05**±0.0 / 1.56±0.0 | **0.03**±0.0 / **0.03**±0.0 / 1.55±0.0 | **0.13**±0.0 / 0.17±0.0 / 0.24±0.0 | **0.13**±0.0 / 0.21±0.0 / 0.29±0.0 | **0.13**±0.0 / 0.75±0.1 / 0.97±0.0 | **0.10**±0.0 / **0.09**±0.0 / **0.14**±0.0 | 0.04±0.0 / **0.03**±0.0 / **0.16**±0.0 |
| | GCN-DE | 0.06±0.0 / 0.06±0.0 / **1.54**±0.0 | 0.03±0.0 / 0.03±0.0 / 1.53±0.0 | 0.14±0.0 / 0.21±0.0 / 0.30±0.0 | 0.14±0.0 / 0.25±0.0 / 0.35±0.0 | 0.14±0.0 / 0.83±0.1 / 1.02±0.1 | 0.11±0.0 / 0.10±0.0 / 0.15±0.0 | 0.05±0.0 / 0.05±0.0 / 0.17±0.0 |
| | GCN-Ens | **0.05**±0.0 / **0.05**±0.0 / 1.56±0.0 | **0.03**±0.0 / **0.03**±0.0 / 1.55±0.0 | **0.12**±0.0 / 0.17±0.0 / 0.24±0.0 | **0.12**±0.0 / 0.21±0.0 / 0.28±0.0 | **0.12**±0.0 / 0.73±0.1 / 0.93±0.0 | **0.10**±0.0 / **0.09**±0.0 / **0.14**±0.0 | 0.04±0.0 / **0.03**±0.0 / **0.15**±0.0 |
| | GCN-MCD+DE | 0.06±0.0 / **0.06**±0.0 / 1.55±0.0 | 0.03±0.0 / 0.03±0.0 / **1.53**±0.0 | 0.14±0.0 / 0.21±0.0 / 0.30±0.0 | 0.14±0.0 / 0.24±0.0 / 0.34±0.0 | 0.14±0.0 / 0.81±0.0 / 1.00±0.0 | 0.11±0.0 / 0.10±0.0 / 0.15±0.0 | 0.05±0.0 / 0.05±0.0 / 0.17±0.0 |
| | SGCN | 0.16±0.0 / 0.16±0.0 / **1.38**±0.0 | 0.13±0.0 / 0.14±0.0 / **1.36**±0.0 | 0.34±0.0 / 0.52±0.0 / 0.57±0.0 | 0.34±0.0 / 0.56±0.0 / 0.61±0.0 | 0.34±0.0 / **0.72**±0.0 / **0.82**±0.0 | 0.29±0.0 / 0.26±0.0 / 0.43±0.0 | 0.28±0.0 / 0.31±0.0 / 0.39±0.0 |
| | GPN | 0.07±0.0 / 0.07±0.0 / 1.56±0.0 | 0.04±0.0 / 0.04±0.0 / 1.68±0.0 | 0.16±0.0 / **0.16**±0.0 / **0.19**±0.0 | 0.16±0.0 / **0.15**±0.0 / **0.17**±0.0 | 0.16±0.0 / **0.15**±0.0 / **0.17**±0.0 | 0.14±0.0 / 0.13±0.0 / 0.17±0.0 | **0.02**±0.0 / 0.18±0.2 / 0.36±0.2 |
| | GCN (Ours) | **0.05**±0.0 / **0.05**±0.0 / 1.56±0.0 | **0.03**±0.0 / **0.03**±0.0 / 1.55±0.0 | **0.12**±0.0 / 0.17±0.0 / 0.24±0.0 | **0.12**±0.0 / **0.21**±0.0 / **0.28**±0.0 | **0.12**±0.0 / 0.73±0.0 / 0.94±0.0 | **0.10**±0.0 / **0.09**±0.0 / 0.14±0.0 | 0.04±0.0 / **0.03**±0.0 / **0.16**±0.0 |
| Coauthor-CS | BGCN | 0.14±0.0 / **0.14**±0.0 / 1.55±0.0 | 0.08±0.0 / **0.08**±0.0 / 1.58±0.0 | 0.17±0.0 / **0.17**±0.0 / **0.28**±0.0 | 0.17±0.0 / 0.39±0.0 / 0.63±0.1 | 0.17±0.0 / 1.02±0.1 / 1.12±0.1 | 0.15±0.0 / 0.13±0.0 / **0.14**±0.0 | **0.03**±0.0 / **0.03**±0.0 / **0.23**±0.0 |
| | GCN-MCD | **0.13**±0.0 / **0.13**±0.0 / 1.34±0.0 | **0.08**±0.0 / **0.08**±0.0 / 1.37±0.0 | **0.16**±0.0 / 0.28±0.0 / 0.43±0.0 | **0.16**±0.0 / 0.31±0.0 / 0.49±0.0 | **0.16**±0.0 / 1.02±0.0 / 1.32±0.0 | **0.14**±0.0 / **0.12**±0.0 / 0.16±0.0 | 0.04±0.0 / **0.05**±0.0 / 0.23±0.0 |
| | GCN-DE | 0.15±0.0 / 0.16±0.0 / **1.30**±0.0 | 0.10±0.0 / 0.10±0.0 / 1.33±0.0 | 0.20±0.0 / 0.34±0.0 / 0.53±0.0 | 0.20±0.0 / 0.40±0.0 / 0.60±0.0 | 0.20±0.0 / 0.97±0.0 / 1.20±0.0 | 0.17±0.0 / 0.15±0.0 / 0.19±0.0 | 0.08±0.0 / 0.09±0.0 / 0.28±0.0 |
| | GCN-Ens | **0.13**±0.0 / **0.13**±0.0 / 1.33±0.0 | **0.08**±0.0 / **0.08**±0.0 / 1.37±0.0 | **0.16**±0.0 / **0.27**±0.0 / 0.42±0.0 | **0.16**±0.0 / **0.31**±0.0 / **0.48**±0.0 | **0.16**±0.0 / 0.95±0.0 / 1.21±0.0 | **0.14**±0.0 / **0.12**±0.0 / **0.15**±0.0 | 0.04±0.0 / 0.05±0.0 / **0.23**±0.0 |
| | GCN-MCD+DE | 0.15±0.0 / 0.16±0.0 / 1.30±0.0 | 0.10±0.0 / 0.10±0.0 / 1.33±0.0 | 0.20±0.0 / 0.35±0.0 / 0.53±0.0 | 0.20±0.0 / 0.40±0.0 / 0.60±0.0 | 0.20±0.0 / 0.99±0.0 / 1.22±0.0 | 0.17±0.0 / 0.15±0.0 / 0.18±0.0 | 0.07±0.0 / 0.09±0.0 / 0.27±0.0 |
| | SGCN | 0.90±0.0 / 0.90±0.0 / **1.10**±0.0 | 0.90±0.0 / 0.90±0.0 / **1.10**±0.0 | 0.93±0.0 / 0.93±0.0 / 0.93±0.0 | 0.93±0.0 / 0.93±0.0 / 0.93±0.0 | 0.93±0.0 / **0.93**±0.0 / **0.93**±0.0 | 0.93±0.0 / 0.93±0.0 / 0.93±0.0 | 0.93±0.0 / 0.93±0.0 / 0.93±0.0 |
| | GPN | 0.24±0.0 / 0.25±0.0 / 1.34±0.0 | 0.18±0.0 / 0.19±0.0 / **1.31**±0.0 | 0.30±0.0 / 0.31±0.0 / **0.36**±0.0 | 0.30±0.0 / **0.29**±0.0 / **0.34**±0.0 | 0.30±0.0 / **0.29**±0.0 / **0.34**±0.0 | 0.31±0.0 / 0.28±0.0 / 0.29±0.0 | **0.03**±0.0 / 0.33±0.1 / 0.58±0.1 |
| | GCN (Ours) | **0.13**±0.0 / **0.13**±0.0 / 1.33±0.0 | **0.08**±0.0 / **0.08**±0.0 / 1.37±0.0 | **0.16**±0.0 / 0.28±0.0 / 0.43±0.0 | **0.16**±0.0 / 0.32±0.0 / 0.49±0.0 | **0.16**±0.0 / 1.01±0.0 / 1.32±0.0 | **0.14**±0.0 / **0.12**±0.0 / 0.15±0.0 | 0.04±0.0 / 0.05±0.0 / 0.24±0.0 |

Table 22: Brier score (↓) for different backbones on clean data as well as i.d. and o.o.d. nodes after a distribution shift in an inductive setting (best and runner-up).

| | Model | LoC (*last*) Clean / I.d. / O.o.D. | LoC (*heterophilic*) Clean / I.d. / O.o.D. | Ber($\hat{p}$) (*near*) Clean / I.d. / O.o.D. | Ber(0.5) Clean / I.d. / O.o.D. | $\mathcal{N}(0,1)$ (*far*) Clean / I.d. / O.o.D. | Page Rank Clean / I.d. / O.o.D. | Homophily Clean / I.d. / O.o.D. |
|---|---|---|---|---|---|---|---|---|
| CoraML | BGCN | 0.17±0.0 / 0.17±0.0 / 1.66±0.0 | 0.27±0.1 / 0.27±0.1 / 1.66±0.0 | 0.29±0.0 / 0.33±0.0 / 0.46±0.0 | 0.29±0.0 / 0.47±0.1 / 0.71±0.2 | 0.29±0.0 / 1.23±0.1 / 1.38±0.1 | 0.31±0.0 / 0.31±0.0 / 0.33±0.0 | 0.10±0.0 / 0.10±0.0 / 0.46±0.0 |
| | GCN-MCD | 0.15±0.0 / 0.15±0.0 / 1.45±0.0 | 0.22±0.0 / 0.22±0.0 / 1.42±0.0 | 0.27±0.0 / 0.36±0.0 / 0.50±0.0 | 0.27±0.0 / 0.36±0.0 / 0.52±0.0 | 0.27±0.0 / 1.17±0.1 / 1.43±0.0 | 0.27±0.0 / 0.27±0.0 / 0.32±0.0 | 0.10±0.0 / 0.10±0.0 / 0.41±0.0 |
| | GCN-DE | 0.21±0.0 / 0.21±0.0 / 1.39±0.0 | 0.29±0.0 / 0.29±0.0 / 1.37±0.0 | 0.34±0.0 / 0.45±0.0 / 0.63±0.0 | 0.34±0.0 / 0.48±0.0 / 0.65±0.0 | 0.34±0.0 / 1.05±0.0 / 1.35±0.0 | 0.32±0.0 / 0.32±0.0 / 0.39±0.0 | 0.20±0.0 / 0.20±0.0 / 0.47±0.0 |
| | GCN-Ens | 0.15±0.0 / 0.15±0.0 / 1.45±0.0 | 0.22±0.0 / 0.22±0.0 / 1.42±0.0 | 0.27±0.0 / 0.36±0.0 / 0.50±0.0 | 0.27±0.0 / 0.37±0.0 / 0.51±0.0 | 0.27±0.0 / 1.12±0.1 / 1.39±0.1 | 0.27±0.0 / 0.27±0.0 / 0.32±0.0 | 0.10±0.0 / 0.10±0.0 / 0.41±0.0 |
| | GCN-MCD+DE | 0.20±0.0 / 0.20±0.0 / 1.39±0.0 | 0.30±0.0 / 0.29±0.0 / 1.38±0.0 | 0.35±0.0 / 0.46±0.0 / 0.64±0.0 | 0.35±0.0 / 0.48±0.0 / 0.66±0.0 | 0.35±0.0 / 1.02±0.0 / 1.31±0.0 | 0.32±0.0 / 0.32±0.0 / 0.39±0.0 | 0.20±0.0 / 0.20±0.0 / 0.47±0.0 |
| | SGCN | 0.27±0.0 / 0.27±0.0 / 1.29±0.0 | 0.28±0.0 / 0.28±0.0 / 1.30±0.0 | 0.70±0.2 / 0.77±0.1 / 0.80±0.1 | 0.70±0.2 / 0.77±0.1 / 0.80±0.1 | 0.70±0.2 / 0.88±0.0 / 1.01±0.2 | 0.46±0.0 / 0.46±0.0 / 0.60±0.0 | 0.41±0.0 / 0.41±0.0 / 0.60±0.0 |
| | GPN | 0.16±0.0 / 0.16±0.0 / 1.44±0.0 | 0.23±0.0 / 0.23±0.0 / 1.41±0.0 | 0.32±0.0 / 0.33±0.0 / 0.39±0.0 | 0.32±0.0 / 0.34±0.0 / 0.41±0.1 | 0.32±0.0 / 0.32±0.0 / 0.37±0.0 | 0.33±0.0 / 0.33±0.0 / 0.35±0.0 | 0.14±0.0 / 0.14±0.0 / 0.48±0.0 |
| | GCN (Ours) | 0.15±0.0 / 0.15±0.0 / 1.45±0.0 | 0.22±0.0 / 0.22±0.0 / 1.42±0.0 | 0.27±0.0 / 0.36±0.0 / 0.51±0.0 | 0.27±0.0 / 0.36±0.0 / 0.51±0.0 | 0.27±0.0 / 1.19±0.0 / 1.46±0.1 | 0.27±0.0 / 0.27±0.0 / 0.32±0.0 | 0.10±0.0 / 0.10±0.0 / 0.41±0.0 |
| CoraML LLM | BGCN | 0.12±0.0 / 0.12±0.0 / 1.61±0.1 | 0.14±0.0 / 0.14±0.0 / 1.61±0.1 | n.a. / n.a. / n.a. | 0.25±0.0 / 0.30±0.0 / 0.39±0.0 | 0.25±0.0 / 0.77±0.0 / 1.09±0.0 | 0.23±0.0 / 0.23±0.0 / 0.26±0.0 | 0.08±0.0 / 0.08±0.0 / 0.44±0.0 |
| | GCN-MCD | 0.12±0.0 / 0.12±0.0 / 1.53±0.0 | 0.14±0.0 / 0.14±0.0 / 1.53±0.1 | n.a. / n.a. / n.a. | 0.25±0.0 / 0.31±0.0 / 0.42±0.0 | 0.25±0.0 / 0.76±0.0 / 1.14±0.0 | 0.22±0.0 / 0.22±0.0 / 0.26±0.0 | 0.09±0.0 / 0.09±0.0 / 0.42±0.0 |
| | GCN-DE | 0.15±0.0 / 0.15±0.0 / 1.49±0.0 | 0.17±0.0 / 0.17±0.0 / 1.50±0.1 | n.a. / n.a. / n.a. | 0.29±0.0 / 0.36±0.0 / 0.52±0.0 | 0.29±0.0 / 0.73±0.0 / 1.09±0.0 | 0.26±0.0 / 0.26±0.0 / 0.31±0.0 | 0.12±0.0 / 0.12±0.0 / 0.44±0.0 |
| | GCN-Ens | 0.12±0.0 / 0.12±0.0 / 1.53±0.0 | 0.14±0.0 / 0.14±0.0 / 1.53±0.1 | n.a. / n.a. / n.a. | 0.25±0.0 / 0.31±0.0 / 0.41±0.0 | 0.25±0.0 / 0.71±0.1 / 1.13±0.0 | 0.22±0.0 / 0.22±0.0 / 0.26±0.0 | 0.08±0.0 / 0.08±0.0 / 0.42±0.0 |
| | GCN-MCD+DE | 0.15±0.0 / 0.15±0.0 / 1.49±0.0 | 0.17±0.0 / 0.17±0.0 / 1.49±0.0 | n.a. / n.a. / n.a. | 0.29±0.0 / 0.36±0.0 / 0.53±0.0 | 0.29±0.0 / 0.74±0.0 / 1.11±0.0 | 0.25±0.0 / 0.25±0.0 / 0.31±0.0 | 0.13±0.0 / 0.13±0.0 / 0.45±0.0 |
| | SGCN | 0.21±0.0 / 0.21±0.0 / 1.31±0.0 | 0.24±0.0 / 0.24±0.0 / 1.31±0.0 | n.a. / n.a. / n.a. | 0.60±0.2 / 0.70±0.1 / 0.75±0.1 | 0.60±0.2 / 0.78±0.0 / 0.88±0.0 | 0.42±0.0 / 0.42±0.0 / 0.56±0.0 | 0.34±0.0 / 0.34±0.0 / 0.55±0.0 |
| | GPN | 0.14±0.0 / 0.14±0.0 / 1.45±0.0 | 0.18±0.0 / 0.18±0.0 / 1.43±0.0 | n.a. / n.a. / n.a. | 0.30±0.0 / 0.31±0.0 / 0.32±0.0 | 0.30±0.0 / 0.30±0.0 / 0.32±0.0 | 0.28±0.0 / 0.28±0.0 / 0.31±0.0 | 0.12±0.0 / 0.12±0.0 / 0.47±0.0 |
| | GCN (Ours) | 0.12±0.0 / 0.12±0.0 / 1.53±0.0 | 0.14±0.0 / 0.14±0.0 / 1.53±0.1 | n.a. / n.a. / n.a. | 0.25±0.0 / 0.31±0.0 / 0.41±0.0 | 0.25±0.0 / 0.75±0.0 / 1.09±0.1 | 0.22±0.0 / 0.22±0.0 / 0.26±0.0 | 0.09±0.0 / 0.09±0.0 / 0.42±0.0 |
| Citeseer | BGCN | 0.27±0.0 / 0.27±0.0 / 1.55±0.0 | 0.15±0.0 / 0.15±0.0 / 1.61±0.0 | 0.28±0.0 / 0.37±0.0 / 0.58±0.0 | 0.28±0.0 / 0.47±0.1 / 0.82±0.1 | 0.28±0.0 / 0.98±0.0 / 1.25±0.0 | 0.28±0.0 / 0.28±0.0 / 0.23±0.0 | 0.20±0.0 / 0.20±0.0 / 0.27±0.0 |
| | GCN-MCD | 0.24±0.0 / 0.24±0.0 / 1.43±0.0 | 0.14±0.0 / 0.14±0.0 / 1.47±0.0 | 0.27±0.0 / 0.37±0.0 / 0.59±0.0 | 0.27±0.0 / 0.38±0.0 / 0.62±0.1 | 0.27±0.0 / 1.00±0.0 / 1.45±0.0 | 0.26±0.0 / 0.26±0.0 / 0.24±0.0 | 0.20±0.0 / 0.20±0.0 / 0.26±0.0 |
| | GCN-DE | 0.29±0.0 / 0.29±0.0 / 1.39±0.0 | 0.22±0.0 / 0.22±0.0 / 1.41±0.0 | 0.34±0.0 / 0.45±0.0 / 0.71±0.0 | 0.34±0.0 / 0.48±0.0 / 0.76±0.1 | 0.34±0.0 / 0.88±0.0 / 1.31±0.0 | 0.32±0.0 / 0.32±0.0 / 0.33±0.0 | 0.28±0.0 / 0.28±0.0 / 0.33±0.0 |
| | GCN-Ens | 0.24±0.0 / 0.24±0.0 / 1.42±0.0 | 0.14±0.0 / 0.14±0.0 / 1.46±0.0 | 0.27±0.0 / 0.37±0.0 / 0.58±0.0 | 0.27±0.0 / 0.37±0.0 / 0.59±0.1 | 0.27±0.0 / 0.98±0.0 / 1.37±0.0 | 0.26±0.0 / 0.26±0.0 / 0.24±0.0 | 0.20±0.0 / 0.20±0.0 / 0.26±0.0 |
| | GCN-MCD+DE | 0.29±0.0 / 0.29±0.0 / 1.39±0.0 | 0.22±0.0 / 0.22±0.0 / 1.42±0.0 | 0.34±0.0 / 0.45±0.0 / 0.71±0.0 | 0.34±0.0 / 0.47±0.0 / 0.75±0.1 | 0.34±0.0 / 0.89±0.0 / 1.33±0.0 | 0.32±0.0 / 0.32±0.0 / 0.33±0.0 | 0.28±0.0 / 0.28±0.0 / 0.33±0.0 |
| | SGCN | 0.34±0.0 / 0.34±0.0 / 1.28±0.0 | 0.28±0.0 / 0.28±0.0 / 1.29±0.0 | 0.47±0.0 / 0.59±0.0 / 0.71±0.0 | 0.47±0.0 / 0.59±0.0 / 0.71±0.0 | 0.47±0.0 / 0.90±0.0 / 1.24±0.0 | 0.44±0.0 / 0.44±0.0 / 0.52±0.0 | 0.42±0.0 / 0.42±0.0 / 0.44±0.0 |
| | GPN | 0.22±0.0 / 0.22±0.0 / 1.49±0.1 | 0.14±0.0 / 0.14±0.0 / 1.55±0.0 | 0.25±0.0 / 0.30±0.0 / 0.40±0.0 | 0.25±0.0 / 0.31±0.0 / 0.44±0.1 | 0.25±0.0 / 0.27±0.0 / 0.30±0.0 | 0.26±0.0 / 0.26±0.0 / 0.18±0.0 | 0.18±0.0 / 0.18±0.0 / 0.24±0.0 |
| | GCN (Ours) | 0.24±0.0 / 0.24±0.0 / 1.42±0.0 | 0.15±0.0 / 0.15±0.0 / 1.47±0.0 | 0.27±0.0 / 0.37±0.0 / 0.58±0.0 | 0.27±0.0 / 0.39±0.0 / 0.63±0.1 | 0.27±0.0 / 1.03±0.0 / 1.42±0.0 | 0.26±0.0 / 0.26±0.0 / 0.24±0.0 | 0.20±0.0 / 0.20±0.0 / 0.26±0.0 |
| PubMed | BGCN | 0.15±0.0 / 0.15±0.0 / 1.89±0.0 | 0.26±0.0 / 0.26±0.0 / 1.86±0.0 | n.a. / n.a. / n.a. | 0.34±0.0 / 0.39±0.0 / 0.57±0.1 | 0.34±0.0 / 0.87±0.1 / 1.12±0.0 | 0.37±0.1 / 0.37±0.1 / 0.34±0.1 | 0.17±0.0 / 0.17±0.0 / 0.49±0.0 |
| | GCN-MCD | 0.12±0.0 / 0.12±0.0 / 1.76±0.0 | 0.24±0.0 / 0.24±0.0 / 1.69±0.0 | n.a. / n.a. / n.a. | 0.32±0.0 / 0.35±0.0 / 0.48±0.0 | 0.32±0.0 / 0.77±0.0 / 1.06±0.0 | 0.31±0.0 / 0.31±0.0 / 0.31±0.0 | 0.17±0.0 / 0.17±0.0 / 0.45±0.0 |
| | GCN-DE | 0.13±0.0 / 0.13±0.0 / 1.75±0.0 | 0.28±0.0 / 0.28±0.0 / 1.65±0.0 | n.a. / n.a. / n.a. | 0.35±0.0 / 0.40±0.0 / 0.57±0.0 | 0.35±0.0 / 0.73±0.0 / 1.09±0.0 | 0.34±0.0 / 0.34±0.0 / 0.35±0.0 | 0.23±0.0 / 0.23±0.0 / 0.44±0.0 |
| | GCN-Ens | 0.12±0.0 / 0.12±0.0 / 1.76±0.0 | 0.24±0.0 / 0.24±0.0 / 1.69±0.0 | n.a. / n.a. / n.a. | 0.32±0.0 / 0.35±0.0 / 0.48±0.0 | 0.32±0.0 / 0.78±0.0 / 1.07±0.0 | 0.31±0.0 / 0.31±0.0 / 0.31±0.0 | 0.17±0.0 / 0.17±0.0 / 0.44±0.0 |
| | GCN-MCD+DE | 0.14±0.0 / 0.14±0.0 / 1.73±0.0 | 0.28±0.0 / 0.28±0.0 / 1.65±0.0 | n.a. / n.a. / n.a. | 0.35±0.0 / 0.40±0.0 / 0.56±0.0 | 0.35±0.0 / 0.72±0.0 / 1.08±0.0 | 0.34±0.0 / 0.34±0.0 / 0.34±0.0 | 0.22±0.0 / 0.22±0.0 / 0.44±0.0 |
| | SGCN | 0.14±0.0 / 0.14±0.0 / 1.66±0.0 | 0.24±0.0 / 0.24±0.0 / 1.61±0.0 | n.a. / n.a. / n.a. | 0.35±0.0 / 0.41±0.0 / 0.51±0.0 | 0.35±0.0 / 0.67±0.0 / 0.95±0.0 | 0.33±0.0 / 0.33±0.0 / 0.39±0.0 | 0.22±0.0 / 0.22±0.0 / 0.40±0.0 |
| | GPN | 0.12±0.0 / 0.12±0.0 / 1.73±0.0 | 0.21±0.0 / 0.21±0.0 / 1.70±0.0 | n.a. / n.a. / n.a. | 0.30±0.0 / 0.34±0.0 / 0.41±0.1 | 0.30±0.0 / 0.30±0.0 / 0.33±0.0 | 0.29±0.0 / 0.29±0.0 / 0.28±0.0 | 0.15±0.0 / 0.15±0.0 / 0.46±0.0 |
| | GCN (Ours) | 0.12±0.0 / 0.12±0.0 / 1.76±0.0 | 0.24±0.0 / 0.24±0.0 / 1.70±0.0 | n.a. / n.a. / n.a. | 0.32±0.0 / 0.35±0.0 / 0.48±0.0 | 0.32±0.0 / 0.78±0.0 / 1.08±0.0 | 0.31±0.0 / 0.31±0.0 / 0.31±0.0 | 0.17±0.0 / 0.17±0.0 / 0.45±0.0 |
| Amazon Photo | BGCN | 0.11±0.0 / 0.11±0.0 / 1.77±0.1 | 0.06±0.0 / 0.06±0.0 / 1.78±0.1 | 0.15±0.0 / 0.25±0.1 / 0.33±0.1 | 0.15±0.0 / 0.62±0.2 / 0.78±0.3 | 0.15±0.0 / 0.83±0.1 / 0.99±0.0 | 0.14±0.0 / 0.14±0.0 / 0.20±0.0 | 0.04±0.0 / 0.04±0.0 / 0.29±0.1 |
| | GCN-MCD | 0.13±0.0 / 0.13±0.0 / 1.64±0.1 | 0.07±0.0 / 0.07±0.0 / 1.62±0.1 | 0.16±0.0 / 0.26±0.0 / 0.33±0.0 | 0.16±0.0 / 0.51±0.1 / 0.63±0.1 | 0.16±0.0 / 0.81±0.1 / 0.97±0.1 | 0.13±0.0 / 0.13±0.0 / 0.24±0.0 | 0.05±0.0 / 0.05±0.0 / 0.28±0.0 |
| | GCN-DE | 0.14±0.0 / 0.14±0.0 / 1.60±0.0 | 0.08±0.0 / 0.08±0.0 / 1.58±0.1 | 0.18±0.0 / 0.30±0.0 / 0.38±0.0 | 0.18±0.0 / 0.48±0.1 / 0.61±0.1 | 0.18±0.0 / 0.86±0.1 / 1.01±0.1 | 0.14±0.0 / 0.14±0.0 / 0.27±0.0 | 0.07±0.0 / 0.07±0.0 / 0.29±0.0 |
| | GCN-Ens | 0.13±0.0 / 0.13±0.0 / 1.64±0.0 | 0.07±0.0 / 0.07±0.0 / 1.61±0.1 | 0.16±0.0 / 0.26±0.0 / 0.33±0.0 | 0.16±0.0 / 0.50±0.1 / 0.63±0.1 | 0.16±0.0 / 0.81±0.0 / 0.99±0.0 | 0.13±0.0 / 0.13±0.0 / 0.24±0.0 | 0.05±0.0 / 0.05±0.0 / 0.28±0.0 |
| | GCN-MCD+DE | 0.14±0.0 / 0.14±0.0 / 1.61±0.1 | 0.08±0.0 / 0.08±0.0 / 1.57±0.1 | 0.18±0.0 / 0.29±0.0 / 0.38±0.0 | 0.18±0.0 / 0.47±0.1 / 0.60±0.1 | 0.18±0.0 / 0.88±0.1 / 1.01±0.1 | 0.14±0.0 / 0.14±0.0 / 0.26±0.0 | 0.07±0.0 / 0.07±0.0 / 0.30±0.0 |
| | SGCN | 0.26±0.0 / 0.26±0.0 / 1.31±0.0 | 0.23±0.0 / 0.23±0.0 / 1.31±0.0 | 0.88±0.0 / 0.88±0.0 / 0.88±0.0 | 0.88±0.0 / 0.88±0.0 / 0.88±0.0 | 0.88±0.0 / 0.88±0.0 / 0.88±0.0 | 0.53±0.2 / 0.53±0.2 / 0.70±0.1 | 0.88±0.0 / 0.88±0.0 / 0.88±0.0 |
| | GPN | 0.19±0.0 / 0.19±0.0 / 1.45±0.0 | 0.09±0.0 / 0.09±0.0 / 1.52±0.0 | 0.27±0.0 / 0.28±0.0 / 0.32±0.0 | 0.27±0.0 / 0.28±0.0 / 0.32±0.0 | 0.27±0.0 / 0.27±0.0 / 0.31±0.0 | 0.25±0.0 / 0.25±0.0 / 0.28±0.0 | 0.14±0.0 / 0.14±0.0 / 0.37±0.0 |
| | GCN (Ours) | 0.13±0.0 / 0.13±0.0 / 1.65±0.0 | 0.07±0.0 / 0.07±0.0 / 1.61±0.1 | 0.16±0.0 / 0.26±0.0 / 0.33±0.0 | 0.16±0.0 / 0.50±0.1 / 0.63±0.1 | 0.16±0.0 / 0.81±0.1 / 0.98±0.1 | 0.13±0.0 / 0.13±0.0 / 0.24±0.0 | 0.05±0.0 / 0.05±0.0 / 0.28±0.0 |
| Amazon Computers | BGCN | 0.23±0.1 / 0.23±0.1 / 1.67±0.1 | 0.12±0.0 / 0.12±0.0 / 1.66±0.1 | 0.34±0.0 / 0.42±0.1 / 0.48±0.1 | 0.34±0.0 / 0.71±0.2 / 0.83±0.2 | 0.34±0.0 / 0.90±0.0 / 1.05±0.0 | 0.29±0.0 / 0.29±0.0 / 0.30±0.0 | 0.14±0.0 / 0.14±0.0 / 0.45±0.0 |
| | GCN-MCD | 0.21±0.0 / 0.21±0.0 / 1.49±0.1 | 0.12±0.0 / 0.12±0.0 / 1.48±0.1 | 0.29±0.0 / 0.39±0.0 / 0.44±0.0 | 0.29±0.0 / 0.57±0.1 / 0.66±0.1 | 0.29±0.0 / 0.86±0.0 / 1.01±0.1 | 0.27±0.0 / 0.27±0.0 / 0.35±0.0 | 0.14±0.0 / 0.14±0.0 / 0.39±0.0 |
| | GCN-DE | 0.22±0.0 / 0.22±0.0 / 1.46±0.1 | 0.13±0.0 / 0.13±0.0 / 1.45±0.1 | 0.31±0.0 / 0.42±0.0 / 0.49±0.0 | 0.31±0.0 / 0.56±0.1 / 0.66±0.1 | 0.31±0.0 / 0.91±0.0 / 1.04±0.0 | 0.28±0.0 / 0.28±0.0 / 0.37±0.0 | 0.16±0.0 / 0.17±0.0 / 0.41±0.0 |
| | GCN-Ens | 0.20±0.0 / 0.20±0.0 / 1.49±0.1 | 0.12±0.0 / 0.12±0.0 / 1.49±0.1 | 0.29±0.0 / 0.39±0.0 / 0.45±0.0 | 0.29±0.0 / 0.57±0.1 / 0.66±0.1 | 0.29±0.0 / 0.89±0.0 / 1.00±0.0 | 0.27±0.0 / 0.27±0.0 / 0.35±0.0 | 0.14±0.0 / 0.14±0.0 / 0.39±0.0 |
| | GCN-MCD+DE | 0.22±0.0 / 0.22±0.0 / 1.45±0.1 | 0.13±0.0 / 0.13±0.0 / 1.44±0.1 | 0.31±0.0 / 0.42±0.0 / 0.49±0.0 | 0.31±0.0 / 0.57±0.1 / 0.66±0.1 | 0.31±0.0 / 0.89±0.0 / 1.02±0.0 | 0.28±0.0 / 0.28±0.0 / 0.37±0.0 | 0.16±0.0 / 0.16±0.0 / 0.40±0.0 |
| | SGCN | 0.64±0.2 / 0.64±0.2 / 1.19±0.0 | 0.30±0.1 / 0.30±0.1 / 1.26±0.0 | 0.90±0.0 / 0.90±0.0 / 0.90±0.0 | 0.90±0.0 / 0.90±0.0 / 0.90±0.0 | 0.90±0.0 / 0.90±0.0 / 0.90±0.0 | 0.90±0.0 / 0.90±0.0 / 0.90±0.0 | 0.90±0.0 / 0.90±0.0 / 0.90±0.0 |
| | GPN | 0.27±0.0 / 0.27±0.0 / 1.29±0.0 | 0.12±0.0 / 0.12±0.0 / 1.44±0.0 | 0.36±0.0 / 0.38±0.0 / 0.42±0.0 | 0.36±0.0 / 0.37±0.0 / 0.42±0.0 | 0.36±0.0 / 0.36±0.0 / 0.40±0.0 | 0.35±0.0 / 0.35±0.0 / 0.36±0.0 | 0.24±0.1 / 0.24±0.1 / 0.47±0.1 |
| | GCN (Ours) | 0.20±0.0 / 0.20±0.0 / 1.49±0.1 | 0.12±0.0 / 0.12±0.0 / 1.49±0.1 | 0.29±0.0 / 0.39±0.0 / 0.44±0.0 | 0.29±0.0 / 0.56±0.1 / 0.65±0.1 | 0.29±0.0 / 0.86±0.1 / 1.01±0.1 | 0.27±0.0 / 0.27±0.0 / 0.36±0.0 | 0.14±0.0 / 0.14±0.0 / 0.39±0.0 |
| Coauthor Physics | BGCN | 0.05±0.0 / 0.05±0.0 / 1.72±0.0 | 0.03±0.0 / 0.03±0.0 / 1.76±0.1 | 0.12±0.0 / 0.12±0.0 / 0.18±0.0 | 0.12±0.0 / 0.41±0.2 / 0.58±0.2 | 0.12±0.0 / 0.84±0.1 / 0.88±0.1 | 0.09±0.0 / 0.09±0.0 / 0.13±0.0 | 0.02±0.0 / 0.02±0.0 / 0.16±0.0 |
| | GCN-MCD | 0.06±0.0 / 0.06±0.0 / 1.59±0.0 | 0.03±0.0 / 0.03±0.0 / 1.60±0.0 | 0.11±0.0 / 0.15±0.0 / 0.23±0.0 | 0.11±0.0 / 0.20±0.0 / 0.28±0.0 | 0.11±0.0 / 0.72±0.0 / 0.95±0.0 | 0.08±0.0 / 0.08±0.0 / 0.14±0.0 | 0.03±0.0 / 0.03±0.0 / 0.15±0.0 |
| | GCN-DE | 0.06±0.0 / 0.06±0.0 / 1.58±0.0 | 0.04±0.0 / 0.04±0.0 / 1.58±0.0 | 0.12±0.0 / 0.18±0.0 / 0.28±0.0 | 0.12±0.0 / 0.22±0.0 / 0.33±0.0 | 0.12±0.0 / 0.80±0.0 / 1.00±0.0 | 0.09±0.0 / 0.09±0.0 / 0.15±0.0 | 0.04±0.0 / 0.04±0.0 / 0.16±0.0 |
| | GCN-Ens | 0.06±0.0 / 0.06±0.0 / 1.59±0.0 | 0.03±0.0 / 0.03±0.0 / 1.60±0.0 | 0.11±0.0 / 0.15±0.0 / 0.22±0.0 | 0.11±0.0 / 0.20±0.0 / 0.28±0.0 | 0.11±0.0 / 0.73±0.0 / 0.94±0.0 | 0.08±0.0 / 0.08±0.0 / 0.14±0.0 | 0.03±0.0 / 0.03±0.0 / 0.15±0.0 |
| | GCN-MCD+DE | 0.06±0.0 / 0.06±0.0 / 1.58±0.0 | 0.04±0.0 / 0.04±0.0 / 1.58±0.0 | 0.12±0.0 / 0.18±0.0 / 0.28±0.0 | 0.12±0.0 / 0.23±0.0 / 0.34±0.0 | 0.12±0.0 / 0.81±0.0 / 1.01±0.0 | 0.09±0.0 / 0.09±0.0 / 0.15±0.0 | 0.04±0.0 / 0.04±0.0 / 0.16±0.0 |
| | SGCN | 0.15±0.0 / 0.15±0.0 / 1.39±0.0 | 0.13±0.0 / 0.13±0.0 / 1.38±0.0 | 0.32±0.0 / 0.49±0.0 / 0.55±0.0 | 0.32±0.0 / 0.54±0.0 / 0.59±0.0 | 0.32±0.0 / 0.70±0.0 / 0.80±0.0 | 0.24±0.0 / 0.24±0.0 / 0.42±0.0 | 0.24±0.0 / 0.24±0.0 / 0.33±0.0 |
| | GPN | 0.07±0.0 / 0.07±0.0 / 1.54±0.0 | 0.04±0.0 / 0.04±0.0 / 1.67±0.0 | 0.14±0.0 / 0.15±0.0 / 0.17±0.0 | 0.14±0.0 / 0.14±0.0 / 0.16±0.0 | 0.14±0.0 / 0.14±0.0 / 0.16±0.0 | 0.12±0.0 / 0.12±0.0 / 0.17±0.0 | 0.06±0.0 / 0.06±0.0 / 0.22±0.0 |
| | GCN (Ours) | 0.06±0.0 / 0.06±0.0 / 1.59±0.0 | 0.03±0.0 / 0.03±0.0 / 1.60±0.0 | 0.11±0.0 / 0.15±0.0 / 0.22±0.0 | 0.11±0.0 / 0.20±0.0 / 0.28±0.0 | 0.11±0.0 / 0.73±0.0 / 0.95±0.0 | 0.08±0.0 / 0.08±0.0 / 0.14±0.0 | 0.03±0.0 / 0.03±0.0 / 0.15±0.0 |
| Coauthor-CS | BGCN | 0.12±0.0 / 0.12±0.0 / 1.59±0.1 | 0.09±0.0 / 0.09±0.0 / 1.62±0.0 | 0.13±0.0 / 0.16±0.0 / 0.27±0.0 | 0.13±0.0 / 0.43±0.1 / 0.69±0.2 | 0.13±0.0 / 1.01±0.1 / 1.09±0.1 | 0.13±0.0 / 0.13±0.0 / 0.15±0.0 | 0.02±0.0 / 0.02±0.0 / 0.22±0.0 |
| | GCN-MCD | 0.11±0.0 / 0.11±0.0 / 1.39±0.0 | 0.08±0.0 / 0.08±0.0 / 1.43±0.0 | 0.13±0.0 / 0.22±0.0 / 0.37±0.0 | 0.13±0.0 / 0.26±0.0 / 0.44±0.0 | 0.13±0.0 / 1.04±0.0 / 1.36±0.0 | 0.12±0.0 / 0.12±0.0 / 0.17±0.0 | 0.03±0.0 / 0.03±0.0 / 0.22±0.0 |
| | GCN-DE | 0.14±0.0 / 0.14±0.0 / 1.35±0.0 | 0.10±0.0 / 0.11±0.0 / 1.38±0.0 | 0.16±0.0 / 0.29±0.0 / 0.48±0.0 | 0.16±0.0 / 0.35±0.0 / 0.57±0.0 | 0.16±0.0 / 0.99±0.0 / 1.24±0.0 | 0.15±0.0 / 0.15±0.0 / 0.19±0.0 | 0.05±0.0 / 0.05±0.0 / 0.25±0.0 |
| | GCN-Ens | 0.11±0.0 / 0.11±0.0 / 1.39±0.0 | 0.08±0.0 / 0.08±0.0 / 1.43±0.0 | 0.13±0.0 / 0.22±0.0 / 0.37±0.0 | 0.13±0.0 / 0.26±0.0 / 0.44±0.0 | 0.13±0.0 / 0.97±0.0 / 1.25±0.0 | 0.12±0.0 / 0.12±0.0 / 0.16±0.0 | 0.03±0.0 / 0.03±0.0 / 0.21±0.0 |
| | GCN-MCD+DE | 0.14±0.0 / 0.14±0.0 / 1.34±0.0 | 0.11±0.0 / 0.11±0.0 / 1.38±0.0 | 0.16±0.0 / 0.29±0.0 / 0.47±0.0 | 0.16±0.0 / 0.34±0.0 / 0.56±0.0 | 0.16±0.0 / 0.97±0.0 / 1.22±0.0 | 0.15±0.0 / 0.15±0.0 / 0.19±0.0 | 0.05±0.0 / 0.05±0.0 / 0.24±0.0 |
| | SGCN | 0.88±0.1 / 0.88±0.1 / 1.10±0.0 | 0.90±0.0 / 0.90±0.0 / 1.10±0.0 | 0.93±0.0 / 0.93±0.0 / 0.93±0.0 | 0.93±0.0 / 0.93±0.0 / 0.93±0.0 | 0.93±0.0 / 0.93±0.0 / 0.93±0.0 | 0.93±0.0 / 0.93±0.0 / 0.93±0.0 | 0.93±0.0 / 0.93±0.0 / 0.93±0.0 |
| | GPN | 0.25±0.0 / 0.25±0.0 / 1.34±0.0 | 0.19±0.0 / 0.19±0.0 / 1.30±0.0 | 0.28±0.0 / 0.28±0.0 / 0.34±0.0 | 0.28±0.0 / 0.27±0.0 / 0.32±0.0 | 0.28±0.0 / 0.27±0.0 / 0.32±0.0 | 0.27±0.0 / 0.27±0.0 / 0.28±0.0 | 0.15±0.0 / 0.15±0.0 / 0.39±0.0 |
| | GCN (Ours) | 0.11±0.0 / 0.11±0.0 / 1.40±0.0 | 0.08±0.0 / 0.08±0.0 / 1.43±0.0 | 0.13±0.0 / 0.22±0.0 / 0.37±0.0 | 0.13±0.0 / 0.27±0.0 / 0.45±0.0 | 0.13±0.0 / 1.04±0.0 / 1.35±0.0 | 0.12±0.0 / 0.12±0.0 / 0.16±0.0 | 0.03±0.0 / 0.03±0.0 / 0.22±0.0 |

Table 23: Brier Score (↓) for different backbones on clean data as well as i.d. and o.o.d. nodes after a distribution shift in a transductive setting (best and runner-up).

| | Model | LoC (*last*) | LoC (*hetero*) | Ber(0.5) | Ber($\hat{p}$) (*near*) | $\mathcal{N}(\mathbf{0}, \boldsymbol{I})$ (*far*) | Homophily | Page Rank |
|---|---|---|---|---|---|---|---|---|
| **GCN** | EBM | 89.9±0.9 | 88.4±2.3 | 58.7±4.6 | 67.1±2.0 | 26.4±1.2 | 71.2±1.6 | **74.2**±2.1 |
| | Safe | **91.6**±0.7 | **90.5**±1.8 | 53.5±3.3 | 56.9±1.7 | 36.1±2.2 | 73.1±1.5 | 50.6±0.6 |
| | GEBM | **91.6**±0.8 | 89.5±1.7 | **94.3**±2.4 | **77.1**±1.8 | **86.6**±3.9 | **76.7**±1.3 | 57.7±1.3 |
| **GAT** | EBM | 90.2±1.2 | 89.7±1.0 | 58.1±2.7 | 55.6±1.6 | 44.1±2.0 | 72.1±1.6 | 51.4±1.0 |
| | Safe | **91.5**±0.8 | **90.8**±0.9 | 54.6±2.2 | 53.2±1.0 | 45.4±1.6 | 70.3±1.9 | 49.9±1.1 |
| | GEBM | 85.0±5.4 | 83.5±5.6 | **89.6**±5.2 | **69.3**±4.2 | **76.5**±4.5 | **72.6**±2.5 | **53.3**±1.8 |
| **GIN** | EBM | 76.5±6.1 | 71.9±4.9 | 46.6±2.5 | **52.3**±2.4 | 42.8±1.9 | 53.2±1.9 | **69.2**±0.8 |
| | Safe | 79.0±5.0 | **76.4**±3.3 | 47.6±1.2 | 49.3±1.3 | 46.2±1.6 | 51.2±1.8 | 47.7±0.5 |
| | GEBM | **80.7**±4.5 | **76.4**±6.1 | **51.6**±1.9 | 51.4±2.1 | **53.6**±1.7 | **65.4**±1.8 | 52.4±1.3 |
| **SAGE** | EBM | 74.0±6.4 | 75.0±7.1 | 47.1±3.0 | **52.7**±2.5 | 42.2±2.8 | 53.2±3.8 | **68.9**±1.2 |
| | Safe | **77.3**±5.4 | **78.2**±4.7 | 48.1±1.1 | 49.2±1.3 | 46.6±1.5 | 51.1±3.5 | 48.0±0.5 |
| | GEBM | **77.3**±8.2 | 73.3±7.8 | **51.9**±1.2 | 51.6±1.7 | **54.7**±1.3 | **62.7**±3.3 | 53.0±2.1 |

Table 24: O.o.d. detection AUC-ROC(↑) using different backbones on CoraML in an inductive setting.

| | Model | LoC *(last)* | LoC *(hetero)* | Ber(0.5) | Ber($\hat{p}$) *(near)* | $\mathcal{N}(\mathbf{0}, \boldsymbol{I})$ *(far)* | Homophily | Page Rank |
|---|---|---|---|---|---|---|---|---|
| **GCN** | EBM | 85.5±1.8 | 75.1±5.5 | 53.9±3.9 | 63.5±2.3 | 36.1±2.1 | 66.3±2.4 | **71.6**±3.5 |
| | Safe | 87.3±2.2 | **76.5**±5.5 | 51.0±2.4 | 55.3±1.9 | 40.3±2.7 | 63.7±2.1 | 54.0±0.6 |
| | GEBM | **88.0**±1.5 | 75.1±3.3 | **94.3**±2.5 | **70.6**±2.5 | **88.9**±3.4 | **68.8**±2.2 | 56.8±1.1 |
| **GAT** | EBM | 86.4±2.9 | 80.3±2.9 | 56.3±3.8 | 56.5±1.5 | 45.2±2.5 | **67.0**±1.3 | **55.1**±0.8 |
| | Safe | **88.9**±1.4 | **83.1**±3.0 | 53.5±2.9 | 54.2±1.2 | 45.4±2.3 | 62.1±1.7 | 53.3±1.0 |
| | GEBM | 79.5±8.4 | 64.4±12.0 | **88.6**±6.8 | **61.9**±4.7 | **81.8**±4.1 | 66.5±3.0 | 54.7±1.3 |
| **GIN** | EBM | 70.2±8.2 | 49.1±9.9 | 46.0±1.9 | **52.8**±2.6 | 44.0±1.9 | 51.4±3.5 | **65.7**±1.0 |
| | Safe | 72.3±6.4 | 51.4±7.1 | 47.9±1.1 | 50.3±0.9 | 46.6±2.1 | 46.1±2.1 | 50.1±0.4 |
| | GEBM | **75.1**±5.9 | **52.7**±8.2 | **51.2**±2.6 | 51.4±2.6 | **52.9**±2.5 | **54.7**±2.6 | 54.0±2.0 |
| **SAGE** | EBM | 66.6±8.2 | 55.3±13.4 | 46.4±2.4 | **53.5**±3.7 | 43.4±2.8 | 52.5±4.3 | **65.4**±2.0 |
| | Safe | 70.2±6.8 | **56.4**±10.6 | 48.5±1.1 | 49.9±1.3 | 46.9±1.8 | 46.5±2.7 | 50.3±0.8 |
| | GEBM | **70.6**±10.4 | 49.3±10.2 | **51.5**±2.4 | 51.4±2.4 | **53.1**±2.7 | **55.1**±3.3 | 54.5±2.5 |

Table 25: O.o.d. detection AUC-PR(↑) using different backbones on CoraML in an inductive setting.

| | **Model** | LoC *(last)* | LoC *(hetero.)* | Ber($\hat{p}$) *(near)* | Ber(0.5) | $\mathcal{N}(0,1)$ *(far)* | Page Rank | Homophily | Rank(↓) |
|---|---|---|---|---|---|---|---|---|---|
| CoraML | EBM | 89.9±0.9 | 88.4±2.3 | 67.3±1.9 | 58.6±4.6 | 25.7±1.3 | **74.2**±2.1 | 71.2±1.6 | 3.9 |
| | Indep. | 81.8±1.6 | 78.5±1.4 | **87.1**±2.0 | **99.5**±0.7 | **96.3**±3.3 | **63.9**±1.7 | 69.0±1.7 | 3.7 |
| | Local | 87.5±2.0 | 86.9±2.1 | 57.4±1.6 | 57.9±2.6 | 63.1±2.9 | 51.0±0.5 | **81.3**±0.8 | 4.3 |
| | Group | **92.8**±0.6 | **90.2**±1.1 | 55.5±1.0 | 60.9±1.9 | 63.7±2.5 | 49.5±1.1 | 71.8±1.7 | 3.4 |
| | GEBM-$E_\theta$ | **92.2**±0.6 | **89.9**±2.0 | 75.9±1.6 | 72.3±5.7 | 1.2±0.4 | 57.6±1.4 | 74.3±1.6 | **3.2** |
| | GEBM | 91.6±0.8 | 89.5±1.7 | **77.7**±2.0 | **94.5**±2.4 | **86.4**±3.6 | 57.7±1.3 | **76.7**±1.3 | **2.5** |
| CoraML-LLM | EBM | 87.1±0.9 | 87.5±3.3 | n.a. | 70.0±1.6 | 33.0±2.5 | **72.7**±1.4 | 69.7±0.7 | 3.3 |
| | Indep. | 78.9±1.1 | 82.0±2.7 | n.a. | **92.1**±2.9 | **96.0**±2.0 | **53.2**±0.7 | 65.1±1.0 | 3.6 |
| | Local | 84.5±1.2 | 84.9±2.4 | n.a. | 55.9±1.2 | 62.4±2.8 | 50.9±0.4 | **80.6**±1.1 | 3.7 |
| | Group | **91.6**±1.2 | **91.4**±1.3 | n.a. | 58.5±2.4 | 62.4±3.4 | 49.6±0.7 | 70.7±1.5 | 2.9 |
| | GEBM-$E_\theta$ | **89.8**±1.0 | **90.4**±1.7 | n.a. | **88.5**±1.3 | 7.1±1.4 | 51.2±0.6 | 73.1±0.9 | **2.7** |
| | GEBM | 88.4±1.1 | 89.7±2.0 | n.a. | 82.2±3.8 | **85.3**±3.4 | 51.9±0.7 | **75.0**±1.2 | **2.4** |
| Citeseer | EBM | 87.4±2.5 | 88.0±1.9 | 64.7±1.5 | 43.8±6.5 | 23.5±1.6 | **66.4**±2.0 | 52.0±2.1 | 3.9 |
| | Indep. | 78.2±2.7 | 76.5±1.4 | **69.3**±1.1 | **99.6**±0.8 | **75.5**±8.6 | **50.6**±1.0 | 51.9±1.4 | 3.7 |
| | Local | 86.9±2.2 | **90.8**±1.3 | 60.0±1.3 | 68.6±4.5 | 63.0±2.0 | 47.6±2.1 | **55.5**±1.9 | **3.3** |
| | Group | **90.6**±1.6 | **88.8**±1.1 | 55.5±1.3 | 70.4±2.8 | 61.7±3.0 | 46.2±2.0 | 53.0±2.2 | 3.6 |
| | GEBM-$E_\theta$ | **90.0**±2.2 | 87.8±1.7 | 63.9±1.0 | 45.4±10.4 | 5.6±1.0 | 48.2±1.9 | 53.2±2.2 | 4.0 |
| | GEBM | 88.9±2.1 | 88.7±1.4 | **66.0**±1.2 | **96.3**±2.9 | **72.1**±3.9 | 48.3±1.8 | **53.7**±2.0 | **2.5** |
| PubMed | EBM | 65.8±1.7 | 67.8±2.4 | n.a. | 48.6±4.5 | 27.5±1.1 | **72.6**±2.3 | 52.2±1.5 | 3.9 |
| | Indep. | 66.0±0.9 | 59.6±3.4 | n.a. | **88.6**±7.9 | **82.5**±8.3 | **50.9**±1.6 | 54.8±1.0 | 3.2 |
| | Local | 68.3±1.0 | 66.3±4.9 | n.a. | 56.1±3.2 | 61.4±1.2 | 47.7±1.2 | **62.6**±1.6 | 3.4 |
| | Group | **72.8**±2.2 | **71.4**±4.9 | n.a. | 59.5±2.3 | 60.3±1.4 | 50.6±2.0 | 56.4±2.0 | **2.3** |
| | GEBM-$E_\theta$ | 68.5±2.1 | **69.2**±2.6 | n.a. | 52.6±10.1 | 8.1±0.7 | 49.7±1.5 | 58.9±1.1 | 3.2 |
| | GEBM | **70.9**±1.0 | 67.2±5.1 | n.a. | **84.0**±7.4 | **75.7**±4.2 | 49.4±1.6 | **59.0**±1.7 | **2.6** |
| Amazon Photo | EBM | 75.2±5.0 | 78.1±3.2 | 57.8±0.6 | 52.1±1.7 | 42.6±1.3 | **90.3**±0.9 | 60.0±0.8 | 4.1 |
| | Indep. | 75.6±3.7 | 76.3±3.2 | **91.4**±2.2 | **68.6**±10.6 | **95.8**±2.9 | 59.7±1.2 | 61.4±1.7 | 3.4 |
| | Local | 73.1±4.1 | **90.6**±1.4 | 51.4±1.1 | 49.2±1.0 | 57.3±1.3 | 54.0±0.6 | **68.4**±0.8 | 3.9 |
| | Group | **89.9**±5.4 | 82.3±4.1 | 52.8±1.1 | 50.7±1.5 | 57.6±1.2 | 53.5±0.8 | 59.6±2.1 | 4.3 |
| | GEBM-$E_\theta$ | **85.4**±4.2 | 85.5±3.3 | **92.5**±1.8 | **71.7**±8.6 | 0.5±0.2 | **62.0**±0.6 | 65.9±1.1 | **2.6** |
| | GEBM | 83.6±5.2 | **87.5**±2.9 | 84.8±3.1 | 64.2±8.6 | **92.4**±2.8 | 58.0±0.7 | **68.3**±1.1 | **2.7** |
| Amazon Computers | EBM | 75.9±5.1 | 85.5±3.6 | 56.6±0.7 | 54.9±2.1 | 40.1±1.1 | **85.3**±1.4 | 61.5±0.9 | 3.6 |
| | Indep. | 74.4±3.8 | 79.2±2.9 | **90.0**±1.9 | **82.1**±6.8 | **96.1**±7.0 | 51.8±1.9 | 59.2±0.9 | 4.1 |
| | Local | 75.7±3.3 | 85.1±6.1 | 51.8±0.6 | 50.9±1.1 | 57.4±1.4 | **54.0**±0.4 | **69.2**±1.1 | 3.9 |
| | Group | **86.6**±4.9 | **89.2**±4.0 | 53.3±0.6 | 52.3±1.0 | 58.8±1.3 | 51.3±1.6 | 56.1±2.6 | 4.1 |
| | GEBM-$E_\theta$ | **86.7**±3.6 | **89.7**±2.3 | **89.9**±1.3 | **84.3**±5.9 | 0.1±0.1 | 53.0±1.4 | **65.3**±0.8 | **2.2** |
| | GEBM | 81.8±3.5 | 87.9±3.9 | 85.2±2.2 | 75.4±6.2 | **94.1**±6.6 | 52.6±1.3 | 64.4±1.5 | **3.1** |
| Coauthor-CS | EBM | 89.9±1.6 | 94.2±1.1 | 69.6±0.8 | 71.6±1.6 | 28.8±0.7 | **81.9**±0.8 | 60.5±0.7 | 3.8 |
| | Indep. | 90.3±0.9 | 92.1±1.8 | **99.2**±0.3 | **100.0**±0.0 | **92.7**±5.4 | **64.4**±0.9 | 58.2±0.5 | 3.2 |
| | Local | 73.8±2.5 | 87.6±2.1 | 57.0±1.1 | 59.1±1.0 | 62.7±2.3 | 45.4±0.3 | **77.0**±0.9 | 4.9 |
| | Group | **96.0**±1.2 | 95.6±1.4 | 62.3±1.8 | 65.3±1.5 | 63.9±2.0 | 51.4±0.6 | 58.5±1.7 | 3.8 |
| | GEBM-$E_\theta$ | **94.5**±1.4 | **97.5**±0.6 | 95.5±0.4 | 97.5±0.7 | 0.2±0.1 | 59.9±0.8 | 64.7±0.7 | **2.8** |
| | GEBM | 92.8±1.5 | **96.1**±1.3 | **95.6**±1.1 | **99.5**±0.2 | **90.0**±5.4 | 56.5±0.9 | **69.5**±1.0 | **2.5** |
| Coauthor-Physics | EBM | 93.4±1.8 | 97.0±0.6 | 59.4±1.3 | 62.4±1.4 | 39.5±1.0 | **82.0**±1.6 | 56.5±0.6 | 3.6 |
| | Indep. | 89.6±2.6 | 94.6±1.0 | **91.9**±1.5 | **99.9**±0.1 | **73.4**±7.4 | **65.7**±1.2 | 55.7±0.4 | 3.7 |
| | Local | 91.7±1.9 | 96.3±0.6 | 53.4±0.9 | 55.9±0.9 | 58.0±1.1 | 52.3±0.5 | **66.1**±1.2 | 4.5 |
| | Group | **95.5**±1.9 | **98.3**±0.3 | 54.7±1.0 | 57.0±1.0 | 56.7±1.3 | 54.7±0.7 | 56.4±1.0 | 4.1 |
| | GEBM-$E_\theta$ | **96.6**±0.9 | **98.4**±0.4 | 81.2±2.9 | 90.0±1.7 | 3.6±0.6 | 62.2±1.3 | 58.3±0.6 | **2.8** |
| | GEBM | 94.9±1.8 | **98.4**±0.3 | **81.8**±2.5 | **95.3**±1.5 | **72.7**±5.8 | 60.1±1.1 | **61.3**±0.8 | **2.4** |

Table 26: O.o.d-detection AUC-ROC (↑) using different EBMs in an inductive setting.

| Dataset | Model | LoC *(last)* | LoC *(hetero.)* | Ber($\hat{p}$) *(near)* | Ber(0.5) | $\mathcal{N}(0,1)$ *(far)* | Page Rank | Homophily | Rank(↓) |
|---|---|---|---|---|---|---|---|---|---|
| CoraML | EBM | 85.5±1.8 | **75.1**±5.5 | 63.8±3.7 | 53.9±3.9 | 35.9±1.8 | **71.6**±3.5 | 66.3±2.4 | **3.2** |
| | Indep. | 74.7±2.0 | 58.1±2.3 | **82.3**±2.4 | **99.5**±0.7 | **97.9**±1.8 | **62.9**±1.8 | 61.4±2.1 | 3.6 |
| | Local | 78.2±4.3 | 68.8±4.4 | 56.0±2.7 | 57.1±2.4 | 62.5±3.3 | 54.9±0.4 | **77.7**±1.1 | 4.1 |
| | Group | **89.7**±1.2 | 73.1±2.1 | 54.7±0.8 | 60.9±2.0 | 61.7±2.4 | 51.1±0.9 | 61.4±2.2 | 4.3 |
| | GEBM-$E_\theta$ | **88.8**±1.2 | **74.3**±4.7 | 69.2±2.4 | 63.4±6.3 | 30.6±1.5 | 55.9±1.1 | 64.5±2.5 | 3.5 |
| | GEBM | 88.0±1.5 | **75.1**±3.3 | **71.3**±2.9 | **94.6**±2.5 | **88.8**±3.1 | 56.8±1.1 | **68.8**±2.2 | **2.2** |
| CoraML-LLM | EBM | 84.0±2.3 | 75.9±6.5 | n.a. | 65.1±2.7 | 39.2±1.5 | **69.1**±1.8 | 66.5±1.0 | 3.0 |
| | Indep. | 69.6±2.4 | 63.0±5.1 | n.a. | **82.1**±5.4 | **97.5**±1.2 | **54.7**±0.8 | 58.6±1.3 | 3.5 |
| | Local | 75.2±3.7 | 61.9±4.9 | n.a. | 52.1±1.3 | 60.5±3.9 | 54.4±0.4 | **77.6**±1.5 | 3.5 |
| | Group | **88.1**±1.4 | **77.9**±4.1 | n.a. | 56.3±2.0 | 59.5±3.4 | 51.6±0.6 | 59.7±1.8 | 3.3 |
| | GEBM-$E_\theta$ | **87.7**±1.2 | **78.8**±3.9 | n.a. | **83.2**±2.3 | 30.7±0.9 | 52.4±0.5 | 64.5±1.2 | **2.8** |
| | GEBM | 84.7±2.0 | 76.8±5.1 | n.a. | 72.8±4.9 | **87.3**±3.8 | 53.2±0.6 | **67.5**±2.1 | **2.6** |
| Citeseer | EBM | 71.0±3.8 | 76.3±4.3 | 59.0±2.4 | 42.5±3.3 | 34.8±1.4 | **59.1**±2.1 | 51.8±2.1 | 3.6 |
| | Indep. | 51.8±3.2 | 59.4±1.6 | **63.5**±1.7 | **99.6**±0.8 | **85.4**±5.5 | **49.8**±0.8 | 51.3±1.1 | 3.7 |
| | Local | 63.7±6.5 | **82.0**±2.8 | 57.5±2.3 | 67.2±6.0 | 62.5±2.6 | 48.5±1.4 | **55.5**±1.9 | **2.9** |
| | Group | **71.3**±3.6 | 75.7±2.6 | 52.7±1.7 | 70.9±3.9 | 60.5±3.4 | 45.7±1.4 | 52.1±1.8 | 4.0 |
| | GEBM-$E_\theta$ | **74.3**±3.9 | 74.1±3.8 | 55.3±1.9 | 42.9±5.0 | 30.7±1.4 | 46.8±1.4 | 52.1±1.7 | 4.2 |
| | GEBM | 69.4±4.5 | **78.5**±3.0 | **59.3**±2.0 | **96.6**±2.7 | **75.3**±3.9 | 47.2±1.4 | **52.4**±1.8 | **2.7** |
| PubMed | EBM | 53.5±2.1 | 33.0±3.3 | n.a. | 46.1±3.2 | 39.1±0.7 | **66.0**±3.2 | 53.7±1.7 | 3.4 |
| | Indep. | 51.7±1.5 | 24.8±2.7 | n.a. | **89.1**±9.0 | **89.5**±5.0 | **50.5**±1.3 | 53.3±1.2 | 3.2 |
| | Local | 50.4±1.3 | 29.0±4.5 | n.a. | 55.8±3.4 | 61.1±1.5 | 47.5±0.7 | **61.4**±1.7 | 3.8 |
| | Group | **60.6**±3.0 | **40.6**±7.7 | n.a. | 59.1±2.9 | 59.1±1.2 | 50.4±1.7 | 52.3±1.7 | **2.6** |
| | GEBM-$E_\theta$ | **57.2**±3.0 | **39.0**±3.7 | n.a. | 47.7±5.9 | 31.8±0.4 | 49.0±1.2 | 56.0±1.3 | 3.1 |
| | GEBM | 56.7±1.7 | 33.4±6.1 | n.a. | **83.8**±8.8 | **79.8**±3.2 | 49.0±1.2 | **56.1**±1.8 | **2.5** |
| Amazon Photo | EBM | 63.3±7.7 | 51.4±4.2 | 59.0±0.7 | 52.0±1.7 | 45.1±0.7 | **89.0**±1.4 | 62.1±1.3 | 3.8 |
| | Indep. | 65.6±5.3 | 48.9±4.5 | **88.7**±2.8 | **58.9**±9.5 | **97.6**±1.6 | 57.2±1.3 | 60.3±2.2 | 3.4 |
| | Local | 55.6±6.7 | **73.9**±4.5 | 50.8±1.6 | 49.1±1.0 | 56.9±1.0 | 55.0±0.7 | **65.8**±1.2 | 4.0 |
| | Group | **80.6**±7.6 | 49.4±5.3 | 52.5±1.3 | 49.8±1.2 | 56.7±1.1 | 51.2±0.6 | 55.0±1.8 | 4.6 |
| | GEBM-$E_\theta$ | **76.2**±7.1 | 57.7±5.4 | **88.6**±2.8 | **63.0**±9.2 | 30.5±0.6 | **59.0**±0.5 | 63.8±1.7 | **2.7** |
| | GEBM | 74.9±8.6 | **62.1**±7.3 | 77.0±4.0 | 55.7±6.4 | **94.8**±1.9 | 55.5±0.8 | **66.4**±1.8 | **2.6** |
| Amazon Computers | EBM | 53.7±7.3 | **79.1**±4.5 | 56.6±1.1 | 54.2±3.0 | 43.5±1.2 | **81.8**±1.8 | 60.0±1.1 | 3.3 |
| | Indep. | 49.4±4.0 | 63.9±3.6 | **85.9**±2.7 | **73.1**±9.0 | **97.8**±4.1 | 49.5±1.5 | 57.8±0.7 | 3.9 |
| | Local | 48.3±4.2 | 75.3±7.2 | 50.7±1.6 | 50.1±2.0 | 57.0±2.3 | **52.2**±0.5 | **67.3**±1.1 | 4.1 |
| | Group | **70.1**±3.7 | 77.4±5.0 | 53.2±1.6 | 51.8±2.1 | 58.8±2.0 | 49.2±1.1 | 51.8±1.8 | 4.3 |
| | GEBM-$E_\theta$ | **68.4**±4.8 | **81.3**±3.6 | **84.9**±1.9 | **75.9**±8.1 | 30.6±0.9 | 49.9±1.1 | 60.3±0.7 | **2.7** |
| | GEBM | 65.0±5.3 | 77.6±6.2 | 79.5±3.1 | 67.4±7.1 | **96.0**±4.6 | 50.1±1.2 | **61.3**±1.2 | **2.7** |
| Coauthor-CS | EBM | 89.6±1.8 | 78.7±3.3 | 69.4±1.0 | 71.9±2.4 | 37.2±0.5 | **77.8**±1.2 | 60.9±0.8 | 3.6 |
| | Indep. | 90.2±0.8 | 65.7±6.0 | **98.7**±0.3 | **100.0**±0.0 | **95.9**±3.1 | **61.5**±1.0 | 55.0±0.5 | 3.0 |
| | Local | 62.0±2.0 | 50.1±3.9 | 54.1±1.2 | 57.4±1.7 | 61.7±2.3 | 47.5±0.1 | **77.1**±1.0 | 4.9 |
| | Group | **96.3**±0.9 | 80.8±4.1 | 60.8±1.8 | 64.2±1.6 | 62.6±1.7 | 52.8±0.5 | 51.6±1.2 | 3.9 |
| | GEBM-$E_\theta$ | **95.1**±1.1 | **88.6**±2.1 | 92.6±0.9 | 96.0±1.4 | 30.6±0.4 | 58.1±0.7 | 59.7±0.5 | **3.0** |
| | GEBM | 93.4±1.2 | **81.6**±5.1 | **93.9**±1.4 | **99.6**±0.2 | **92.8**±3.9 | 55.2±0.8 | **65.7**±1.0 | **2.5** |
| Coauthor-Physics | EBM | 79.8±3.2 | 92.0±1.4 | 58.8±1.2 | 62.1±1.9 | 43.6±0.9 | **77.3**±2.0 | 56.9±0.8 | 3.4 |
| | Indep. | 63.9±6.9 | 82.5±2.8 | **90.6**±1.8 | **99.9**±0.1 | **83.9**±4.7 | **60.5**±1.2 | 56.3±0.5 | 3.5 |
| | Local | 64.5±4.8 | 84.2±1.8 | 53.0±0.7 | 54.5±0.8 | 57.9±1.3 | 50.9±0.4 | **68.9**±1.1 | 4.5 |
| | Group | **88.2**±2.8 | 95.2±0.7 | 54.0±0.6 | 56.7±0.7 | 56.2±1.6 | 53.5±0.9 | 53.6±0.6 | 4.2 |
| | GEBM-$E_\theta$ | **89.2**±2.4 | **95.3**±0.9 | 74.8±2.8 | 85.4±2.9 | 31.0±0.5 | 58.1±1.3 | 56.8±0.8 | **3.0** |
| | GEBM | 84.7±3.8 | **95.4**±0.8 | **77.1**±2.7 | **94.1**±1.9 | **80.1**±4.4 | 55.9±1.0 | **61.3**±0.8 | **2.3** |

Table 27: O.o.d-detection AUC-PR (↑) using different EBMs in an inductive setting.

| | Model | LoC *(last)* | LoC *(hetero.)* | Ber($\hat{p}$) *(near)* | Ber(0.5) | $\mathcal{N}(0,1)$ *(far)* | Page Rank | Homophily | Rank(↓) |
|---|---|---|---|---|---|---|---|---|---|
| CoraML | EBM | 87.5±1.8 | 85.3±2.0 | 68.7±1.5 | 61.1±4.9 | 24.7±1.6 | **73.0**±1.7 | 67.9±2.0 | 4.1 |
| | Indep. | 79.2±1.4 | 76.9±1.2 | **86.6**±2.1 | **98.5**±1.6 | **99.7**±0.3 | **55.9**±1.1 | 63.9±1.5 | 3.7 |
| | Local | 86.9±2.1 | 85.4±1.5 | 56.5±2.0 | 54.5±2.2 | 62.1±1.5 | 49.9±0.7 | **79.8**±1.0 | 4.3 |
| | Group | **91.7**±1.2 | **89.6**±1.0 | 57.9±1.0 | 61.3±1.4 | 62.8±2.1 | 48.1±1.3 | 68.9±1.5 | 3.3 |
| | GEBM-$E_\theta$ | **90.1**±1.6 | **88.0**±1.2 | **79.6**±2.3 | 76.9±8.5 | 1.4±0.3 | 52.3±1.1 | 71.1±2.3 | **3.1** |
| | GEBM | 89.7±0.9 | 87.9±1.4 | 78.9±2.4 | **92.4**±3.7 | **89.8**±1.7 | 52.7±1.0 | **73.1**±1.5 | **2.6** |
| CoraML-LLM | EBM | 88.2±2.4 | 87.0±2.3 | n.a. | 69.4±2.4 | 34.1±3.4 | **71.9**±2.1 | 69.0±1.5 | 3.5 |
| | Indep. | **89.7**±1.5 | 79.7±2.7 | n.a. | **91.9**±2.6 | **95.3**±2.4 | **51.2**±0.5 | 62.7±2.1 | 3.4 |
| | Local | **89.7**±1.5 | 85.9±2.6 | n.a. | 54.2±1.9 | 63.0±1.9 | **51.2**±0.5 | **79.7**±1.0 | 3.2 |
| | Group | **89.7**±1.5 | **91.0**±1.9 | n.a. | 57.5±2.0 | 62.9±2.1 | **51.2**±0.5 | 67.7±2.0 | 3.2 |
| | GEBM-$E_\theta$ | **91.2**±1.5 | **90.6**±1.8 | n.a. | **88.9**±2.5 | 7.6±0.9 | 51.0±0.7 | 71.9±2.2 | **2.9** |
| | GEBM | **89.7**±1.5 | 89.1±2.6 | n.a. | 81.1±3.6 | **86.0**±2.5 | **51.2**±0.5 | **72.9**±1.6 | **2.6** |
| Citeseer | EBM | 85.7±1.2 | 85.5±2.1 | 65.8±2.0 | 55.0±6.5 | 23.3±1.2 | **66.3**±1.3 | 50.6±1.0 | 4.0 |
| | Indep. | 76.3±1.7 | 74.3±2.4 | **69.2**±0.8 | **100.0**±0.0 | **93.5**±6.2 | **50.6**±1.4 | 49.2±1.1 | 3.7 |
| | Local | 85.7±2.2 | **89.1**±1.9 | 61.0±1.8 | 72.7±1.4 | 66.3±2.1 | 47.4±0.8 | **54.7**±0.5 | **3.2** |
| | Group | **88.7**±2.1 | **87.2**±2.4 | 55.7±1.2 | 73.0±1.3 | 66.2±1.8 | 45.8±0.9 | 51.6±1.4 | 3.4 |
| | GEBM-$E_\theta$ | **88.4**±1.5 | 85.4±2.2 | 64.7±1.2 | 59.8±9.0 | 5.5±0.7 | 47.8±0.9 | 51.3±1.1 | 4.2 |
| | GEBM | 87.1±2.2 | 86.7±2.2 | **66.6**±1.2 | **98.5**±0.8 | **81.1**±2.8 | 48.0±1.1 | **51.9**±1.1 | **2.5** |
| PubMed | EBM | 66.1±3.0 | 62.8±4.5 | n.a. | 58.5±7.5 | 26.4±0.9 | **71.8**±2.7 | 53.4±0.9 | 3.9 |
| | Indep. | 64.0±1.8 | 59.6±2.9 | n.a. | **89.9**±5.5 | **75.2**±11.7 | 50.8±0.8 | 53.5±0.8 | 3.4 |
| | Local | 66.7±3.3 | 65.2±3.7 | n.a. | 57.5±2.3 | 60.9±1.4 | 47.8±0.7 | **61.6**±2.0 | 3.5 |
| | Group | **69.3**±3.8 | **70.4**±3.8 | n.a. | 59.6±2.1 | 58.9±2.0 | **51.0**±0.9 | 53.4±2.4 | **2.5** |
| | GEBM-$E_\theta$ | **68.4**±3.6 | 65.4±5.1 | n.a. | 66.7±10.3 | 8.0±0.4 | 49.4±1.0 | **59.9**±1.0 | 3.1 |
| | GEBM | **68.4**±3.0 | **66.3**±3.7 | n.a. | **84.0**±5.1 | **73.0**±6.2 | 49.6±0.8 | 57.1±1.9 | **2.4** |
| Amazon Photo | EBM | 75.2±4.7 | 75.9±3.7 | 58.5±1.5 | 51.5±1.4 | 41.3±1.6 | **90.5**±0.9 | 56.8±1.4 | 5.1 |
| | Indep. | 75.5±3.8 | 76.2±3.0 | **86.8**±3.1 | **59.7**±8.9 | **93.8**±3.8 | 58.2±1.6 | **62.2**±1.9 | 3.1 |
| | Local | 76.9±7.3 | **91.4**±2.2 | 71.6±16.9 | 56.3±7.4 | 77.5±17.5 | 54.5±0.6 | **62.2**±1.9 | 3.6 |
| | Group | **89.7**±3.4 | 81.5±5.2 | 72.7±15.5 | 57.0±6.8 | 77.5±17.6 | 53.4±1.1 | **62.2**±1.9 | 3.5 |
| | GEBM-$E_\theta$ | **85.5**±5.9 | 82.1±3.2 | **93.6**±0.6 | **66.3**±9.1 | 0.5±0.1 | **60.7**±1.0 | **61.6**±1.8 | **2.9** |
| | GEBM | 84.2±5.3 | **87.4**±3.9 | 83.3±3.9 | 58.4±8.0 | **92.0**±2.7 | 57.8±1.2 | **62.2**±1.9 | **2.8** |
| Amazon Computers | EBM | 75.3±4.3 | 84.5±3.2 | 57.8±1.6 | 54.6±2.3 | 40.8±1.4 | **87.2**±1.5 | 60.6±0.8 | 4.1 |
| | Indep. | 77.3±1.5 | 79.5±1.9 | **89.5**±1.8 | **80.7**±4.5 | **94.2**±8.2 | 50.9±1.5 | 55.9±1.0 | 4.1 |
| | Local | 77.1±4.3 | 89.5±1.9 | 52.2±0.7 | 50.5±0.7 | 57.9±1.5 | **53.7**±0.6 | **68.2**±1.3 | 3.6 |
| | Group | **89.7**±4.2 | **91.1**±1.2 | 53.8±1.5 | 52.9±1.5 | 58.8±1.8 | 51.3±1.0 | 52.4±2.5 | 3.6 |
| | GEBM-$E_\theta$ | **85.1**±3.3 | 89.1±2.7 | **91.7**±1.5 | **81.8**±5.0 | 0.2±0.1 | 52.7±1.1 | **63.7**±0.6 | **2.7** |
| | GEBM | 84.8±2.8 | **90.1**±1.5 | 84.3±2.8 | 72.9±5.3 | **92.0**±7.7 | 52.1±1.1 | 61.9±1.5 | **2.9** |
| Coauthor-CS | EBM | 88.6±1.1 | 91.8±1.9 | 68.7±0.9 | 70.9±1.3 | 29.4±0.8 | **81.5**±1.2 | 59.5±0.3 | 3.8 |
| | Indep. | 89.5±0.9 | 88.9±3.2 | **99.1**±0.2 | **100.0**±0.0 | **97.6**±1.9 | **63.2**±0.9 | 55.6±0.7 | 3.0 |
| | Local | 73.1±1.8 | 85.1±2.9 | 57.6±0.5 | 58.1±0.9 | 63.9±1.6 | 45.6±0.4 | **77.8**±0.7 | 4.9 |
| | Group | **96.5**±1.1 | **93.8**±2.7 | 61.7±1.6 | 64.0±1.6 | 64.5±1.4 | 51.8±0.9 | 54.5±1.9 | 3.8 |
| | GEBM-$E_\theta$ | **94.2**±1.1 | **96.4**±1.0 | 95.3±0.6 | 97.5±0.5 | 0.4±0.1 | 59.1±0.9 | 63.7±0.7 | **2.8** |
| | GEBM | 92.7±1.3 | 93.6±2.9 | **96.0**±0.6 | **99.2**±0.5 | **95.1**±2.0 | 56.0±0.7 | **68.4**±1.1 | **2.7** |
| Coauthor-Physics | EBM | 91.5±1.7 | 95.3±1.6 | 59.7±1.2 | 62.0±1.4 | 39.6±0.8 | **83.0**±1.0 | 56.5±0.7 | 3.8 |
| | Indep. | 88.2±1.8 | 92.9±1.1 | **92.9**±1.3 | **99.5**±0.8 | **81.5**±4.8 | **66.4**±0.4 | 54.9±0.3 | 3.7 |
| | Local | 89.7±1.7 | 93.8±2.2 | 53.9±0.6 | 55.7±1.1 | 58.8±0.8 | 52.4±0.3 | **67.9**±1.5 | 4.5 |
| | Group | **94.8**±2.1 | **97.8**±0.7 | 56.1±0.9 | 57.8±1.1 | 58.3±1.0 | 55.2±0.4 | 56.6±1.3 | 3.6 |
| | GEBM-$E_\theta$ | **95.7**±1.0 | **97.6**±1.1 | 82.5±2.7 | 90.0±2.8 | 4.0±0.6 | 63.0±0.6 | 58.7±0.9 | **2.8** |
| | GEBM | 93.7±1.9 | 97.2±1.2 | **83.8**±2.7 | **93.4**±2.7 | **78.6**±3.7 | 60.9±0.6 | **61.9**±1.0 | **2.7** |

Table 28: O.o.d-detection AUC-ROC (↑) using different EBMs in a transductive setting.

| | Model | LoC *(last)* | LoC *(hetero.)* | Ber($\hat{p}$) *(near)* | Ber(0.5) | $\mathcal{N}(0,1)$ *(far)* | Page Rank | Homophily | Rank(↓) |
|---|---|---|---|---|---|---|---|---|---|
| CoraML | EBM | 83.0$_{\pm 2.1}$ | 69.6$_{\pm 4.3}$ | 66.1$_{\pm 3.1}$ | 56.3$_{\pm 6.2}$ | 35.0$_{\pm 1.4}$ | **70.2**$_{\pm 2.9}$ | 62.5$_{\pm 2.1}$ | **3.6** |
| | Indep. | 72.2$_{\pm 1.0}$ | 55.5$_{\pm 2.3}$ | **82.1**$_{\pm 4.3}$ | **98.1**$_{\pm 2.6}$ | **99.8**$_{\pm 0.2}$ | **56.3**$_{\pm 1.4}$ | 56.6$_{\pm 1.4}$ | 3.7 |
| | Local | 78.0$_{\pm 4.5}$ | 64.7$_{\pm 3.5}$ | 53.8$_{\pm 2.2}$ | 52.4$_{\pm 2.7}$ | 61.9$_{\pm 3.3}$ | 54.1$_{\pm 0.4}$ | **76.0**$_{\pm 1.0}$ | 4.0 |
| | Group | **88.0**$_{\pm 1.9}$ | **72.0**$_{\pm 2.7}$ | 55.5$_{\pm 3.3}$ | 60.0$_{\pm 3.9}$ | 61.7$_{\pm 3.7}$ | 50.5$_{\pm 0.6}$ | 57.6$_{\pm 1.4}$ | 3.8 |
| | GEBM-$E_\theta$ | **86.5**$_{\pm 2.2}$ | 71.6$_{\pm 3.1}$ | **71.7**$_{\pm 3.8}$ | 67.5$_{\pm 10.3}$ | 30.1$_{\pm 1.5}$ | 52.8$_{\pm 0.9}$ | 61.4$_{\pm 2.2}$ | **3.6** |
| | GEBM | 86.1$_{\pm 1.6}$ | **73.2**$_{\pm 2.8}$ | 71.2$_{\pm 4.4}$ | **91.6**$_{\pm 5.2}$ | **91.7**$_{\pm 1.8}$ | 53.3$_{\pm 0.7}$ | **64.3**$_{\pm 1.4}$ | **2.4** |
| CoraML-LLM | EBM | 84.0$_{\pm 4.7}$ | 73.0$_{\pm 4.5}$ | n.a. | 66.0$_{\pm 4.4}$ | 41.0$_{\pm 2.8}$ | **69.6**$_{\pm 2.5}$ | 64.7$_{\pm 1.6}$ | 3.3 |
| | Indep. | **85.9**$_{\pm 2.9}$ | 58.8$_{\pm 5.3}$ | n.a. | **82.6**$_{\pm 4.1}$ | **97.2**$_{\pm 1.4}$ | 52.5$_{\pm 0.4}$ | 55.3$_{\pm 2.5}$ | 3.7 |
| | Local | **85.9**$_{\pm 2.9}$ | 64.0$_{\pm 4.6}$ | n.a. | 52.5$_{\pm 3.7}$ | 61.9$_{\pm 3.0}$ | 52.5$_{\pm 0.4}$ | **77.2**$_{\pm 0.9}$ | 3.3 |
| | Group | **85.9**$_{\pm 2.9}$ | **75.0**$_{\pm 5.3}$ | n.a. | 55.9$_{\pm 3.3}$ | 61.7$_{\pm 3.3}$ | 52.5$_{\pm 0.4}$ | 57.7$_{\pm 1.6}$ | 3.5 |
| | GEBM-$E_\theta$ | **88.3**$_{\pm 3.2}$ | **78.3**$_{\pm 4.9}$ | n.a. | **84.8**$_{\pm 4.2}$ | 31.7$_{\pm 1.5}$ | **52.6**$_{\pm 0.7}$ | 62.6$_{\pm 1.9}$ | **2.1** |
| | GEBM | **85.9**$_{\pm 2.9}$ | 74.6$_{\pm 6.1}$ | n.a. | 72.4$_{\pm 5.3}$ | **88.4**$_{\pm 2.5}$ | 52.5$_{\pm 0.4}$ | **65.2**$_{\pm 1.9}$ | **2.7** |
| Citeseer | EBM | 66.3$_{\pm 4.4}$ | 72.1$_{\pm 4.2}$ | **60.8**$_{\pm 4.4}$ | 50.0$_{\pm 5.0}$ | 35.3$_{\pm 2.5}$ | **60.2**$_{\pm 1.8}$ | 51.3$_{\pm 0.7}$ | 3.4 |
| | Indep. | 48.6$_{\pm 2.2}$ | 57.7$_{\pm 3.5}$ | **62.7**$_{\pm 2.2}$ | **100.0**$_{\pm 0.0}$ | **96.2**$_{\pm 3.6}$ | **50.0**$_{\pm 1.5}$ | 49.6$_{\pm 0.8}$ | 3.7 |
| | Local | 60.4$_{\pm 4.6}$ | **78.7**$_{\pm 3.8}$ | 58.6$_{\pm 3.4}$ | 73.4$_{\pm 3.5}$ | 65.8$_{\pm 2.7}$ | 48.3$_{\pm 0.9}$ | **54.7**$_{\pm 1.1}$ | **2.9** |
| | Group | **67.0**$_{\pm 5.4}$ | 71.9$_{\pm 3.5}$ | 52.8$_{\pm 2.6}$ | 74.4$_{\pm 2.5}$ | 64.5$_{\pm 2.4}$ | 45.7$_{\pm 0.8}$ | **51.4**$_{\pm 0.5}$ | 3.9 |
| | GEBM-$E_\theta$ | **68.8**$_{\pm 4.6}$ | 69.8$_{\pm 3.2}$ | 56.4$_{\pm 2.7}$ | 51.7$_{\pm 5.9}$ | 31.2$_{\pm 2.0}$ | 47.2$_{\pm 0.6}$ | 51.3$_{\pm 0.4}$ | 4.4 |
| | GEBM | 65.8$_{\pm 5.4}$ | **74.1**$_{\pm 3.9}$ | 59.1$_{\pm 2.5}$ | **98.6**$_{\pm 0.8}$ | **83.3**$_{\pm 2.5}$ | 47.6$_{\pm 1.0}$ | **51.4**$_{\pm 0.3}$ | **2.9** |
| PubMed | EBM | 53.9$_{\pm 2.9}$ | 28.6$_{\pm 4.0}$ | n.a. | 53.1$_{\pm 5.2}$ | 38.0$_{\pm 0.4}$ | **64.4**$_{\pm 4.1}$ | 55.1$_{\pm 1.3}$ | 3.2 |
| | Indep. | 50.5$_{\pm 1.8}$ | 24.3$_{\pm 1.6}$ | n.a. | **90.3**$_{\pm 5.6}$ | **84.9**$_{\pm 7.5}$ | **50.1**$_{\pm 0.7}$ | 52.5$_{\pm 0.8}$ | 3.4 |
| | Local | 50.6$_{\pm 2.3}$ | 27.7$_{\pm 2.8}$ | n.a. | 56.4$_{\pm 2.6}$ | 60.2$_{\pm 2.0}$ | 47.2$_{\pm 0.4}$ | **59.6**$_{\pm 1.6}$ | 3.7 |
| | Group | **58.8**$_{\pm 3.6}$ | **38.2**$_{\pm 5.4}$ | n.a. | 58.9$_{\pm 2.7}$ | 57.4$_{\pm 1.8}$ | 49.7$_{\pm 0.8}$ | 50.1$_{\pm 1.7}$ | **2.6** |
| | GEBM-$E_\theta$ | **56.5**$_{\pm 3.7}$ | **34.3**$_{\pm 6.3}$ | n.a. | 57.7$_{\pm 7.5}$ | 31.3$_{\pm 0.4}$ | 48.4$_{\pm 0.6}$ | **57.5**$_{\pm 0.7}$ | **2.9** |
| | GEBM | 55.9$_{\pm 3.2}$ | 30.8$_{\pm 3.5}$ | n.a. | **83.3**$_{\pm 6.2}$ | **77.3**$_{\pm 5.3}$ | 48.4$_{\pm 0.6}$ | 54.3$_{\pm 1.5}$ | **2.9** |
| Amazon Photo | EBM | 64.3$_{\pm 6.3}$ | 49.1$_{\pm 4.3}$ | 59.3$_{\pm 2.5}$ | 52.0$_{\pm 2.1}$ | 44.3$_{\pm 1.2}$ | **89.5**$_{\pm 1.4}$ | **58.9**$_{\pm 1.9}$ | 3.8 |
| | Indep. | 66.4$_{\pm 5.6}$ | 48.3$_{\pm 4.3}$ | **81.1**$_{\pm 5.3}$ | **52.8**$_{\pm 6.4}$ | **95.8**$_{\pm 2.6}$ | 55.8$_{\pm 1.6}$ | **58.6**$_{\pm 2.0}$ | **3.4** |
| | Local | 62.5$_{\pm 9.9}$ | **75.5**$_{\pm 5.9}$ | 67.1$_{\pm 13.6}$ | 52.1$_{\pm 3.3}$ | 79.4$_{\pm 18.3}$ | 55.3$_{\pm 0.7}$ | **58.6**$_{\pm 2.0}$ | 3.9 |
| | Group | **80.0**$_{\pm 6.2}$ | 50.0$_{\pm 8.5}$ | 68.1$_{\pm 12.4}$ | 52.1$_{\pm 3.6}$ | 79.4$_{\pm 18.3}$ | 51.3$_{\pm 0.6}$ | **58.6**$_{\pm 2.0}$ | 3.7 |
| | GEBM-$E_\theta$ | **78.3**$_{\pm 7.1}$ | 52.3$_{\pm 5.4}$ | **90.8**$_{\pm 1.6}$ | **59.9**$_{\pm 8.9}$ | 30.7$_{\pm 0.7}$ | **57.6**$_{\pm 0.8}$ | **58.6**$_{\pm 2.4}$ | **2.7** |
| | GEBM | 77.1$_{\pm 8.1}$ | **63.2**$_{\pm 10.1}$ | 75.7$_{\pm 4.9}$ | 51.9$_{\pm 5.3}$ | **94.4**$_{\pm 1.7}$ | 55.3$_{\pm 0.9}$ | **58.6**$_{\pm 2.0}$ | 3.5 |
| Amazon Computers | EBM | 50.9$_{\pm 6.4}$ | 77.3$_{\pm 4.4}$ | 57.4$_{\pm 2.1}$ | 53.3$_{\pm 2.9}$ | 43.7$_{\pm 1.2}$ | **84.4**$_{\pm 2.5}$ | 59.4$_{\pm 0.9}$ | 3.8 |
| | Indep. | 49.6$_{\pm 2.6}$ | 63.9$_{\pm 2.9}$ | **84.1**$_{\pm 2.0}$ | **70.5**$_{\pm 4.9}$ | **96.7**$_{\pm 4.8}$ | 49.2$_{\pm 1.5}$ | 55.3$_{\pm 0.9}$ | 4.1 |
| | Local | 49.2$_{\pm 4.3}$ | 79.8$_{\pm 3.3}$ | 50.5$_{\pm 1.3}$ | 49.0$_{\pm 1.2}$ | 57.3$_{\pm 1.9}$ | **51.8**$_{\pm 0.5}$ | **67.7**$_{\pm 1.4}$ | 3.9 |
| | Group | **72.8**$_{\pm 3.2}$ | **79.9**$_{\pm 2.2}$ | 53.0$_{\pm 1.7}$ | 51.7$_{\pm 1.6}$ | 58.5$_{\pm 2.2}$ | 48.9$_{\pm 0.7}$ | 49.3$_{\pm 2.2}$ | 4.0 |
| | GEBM-$E_\theta$ | 62.6$_{\pm 6.1}$ | **79.9**$_{\pm 3.6}$ | **87.9**$_{\pm 1.9}$ | **73.6**$_{\pm 5.9}$ | 30.3$_{\pm 0.5}$ | 50.0$_{\pm 0.9}$ | **59.6**$_{\pm 0.7}$ | **2.6** |
| | GEBM | **66.8**$_{\pm 3.5}$ | **80.4**$_{\pm 3.4}$ | 77.4$_{\pm 2.9}$ | 64.4$_{\pm 4.9}$ | **94.7**$_{\pm 4.9}$ | 49.7$_{\pm 1.1}$ | 59.5$_{\pm 1.5}$ | **2.6** |
| Coauthor-CS | EBM | 87.8$_{\pm 1.7}$ | 72.3$_{\pm 4.4}$ | 69.5$_{\pm 1.3}$ | 72.2$_{\pm 1.9}$ | 38.0$_{\pm 0.6}$ | **77.5**$_{\pm 1.6}$ | 60.5$_{\pm 0.4}$ | 3.6 |
| | Indep. | 88.7$_{\pm 1.2}$ | 58.9$_{\pm 7.6}$ | **98.8**$_{\pm 0.3}$ | **99.9**$_{\pm 0.1}$ | **98.7**$_{\pm 1.0}$ | **60.7**$_{\pm 0.8}$ | 53.0$_{\pm 0.6}$ | 3.0 |
| | Local | 61.0$_{\pm 1.7}$ | 46.9$_{\pm 4.1}$ | 55.2$_{\pm 0.5}$ | 56.3$_{\pm 0.8}$ | 63.3$_{\pm 1.8}$ | 47.6$_{\pm 0.2}$ | **77.8**$_{\pm 0.8}$ | 4.9 |
| | Group | **96.5**$_{\pm 0.8}$ | 74.8$_{\pm 8.2}$ | 60.8$_{\pm 1.6}$ | 63.4$_{\pm 1.6}$ | 63.6$_{\pm 1.6}$ | 52.9$_{\pm 0.7}$ | 48.8$_{\pm 1.4}$ | 3.9 |
| | GEBM-$E_\theta$ | **94.7**$_{\pm 0.8}$ | **84.7**$_{\pm 3.7}$ | 92.8$_{\pm 1.0}$ | 96.3$_{\pm 0.8}$ | 31.2$_{\pm 0.5}$ | 57.5$_{\pm 0.7}$ | 59.3$_{\pm 0.7}$ | **3.0** |
| | GEBM | 92.8$_{\pm 0.9}$ | **74.9**$_{\pm 7.5}$ | **94.9**$_{\pm 0.7}$ | **99.2**$_{\pm 0.6}$ | **96.4**$_{\pm 1.4}$ | 54.9$_{\pm 0.6}$ | **64.7**$_{\pm 1.3}$ | **2.5** |
| Coauthor-Physics | EBM | 75.3$_{\pm 3.5}$ | 88.0$_{\pm 4.0}$ | 59.4$_{\pm 1.7}$ | 61.1$_{\pm 1.9}$ | 43.3$_{\pm 0.6}$ | **78.4**$_{\pm 1.4}$ | 57.5$_{\pm 0.6}$ | 3.6 |
| | Indep. | 60.5$_{\pm 4.6}$ | 79.4$_{\pm 2.7}$ | **91.5**$_{\pm 1.6}$ | **99.1**$_{\pm 1.7}$ | **88.9**$_{\pm 2.8}$ | **61.1**$_{\pm 0.6}$ | 55.5$_{\pm 0.4}$ | 3.2 |
| | Local | 57.3$_{\pm 4.8}$ | 74.5$_{\pm 7.7}$ | 53.0$_{\pm 0.7}$ | 53.9$_{\pm 0.9}$ | 58.5$_{\pm 1.0}$ | 51.1$_{\pm 0.1}$ | **70.5**$_{\pm 1.0}$ | 4.8 |
| | Group | **86.4**$_{\pm 3.0}$ | **93.7**$_{\pm 2.0}$ | 54.7$_{\pm 1.4}$ | 56.8$_{\pm 1.5}$ | 57.3$_{\pm 0.9}$ | 54.2$_{\pm 0.5}$ | 53.5$_{\pm 1.0}$ | 3.9 |
| | GEBM-$E_\theta$ | **86.6**$_{\pm 1.8}$ | **93.1**$_{\pm 3.2}$ | 76.3$_{\pm 3.5}$ | 84.4$_{\pm 4.7}$ | 30.7$_{\pm 0.3}$ | 58.8$_{\pm 0.8}$ | 57.6$_{\pm 0.6}$ | **2.8** |
| | GEBM | 80.6$_{\pm 3.7}$ | 91.5$_{\pm 4.3}$ | **78.6**$_{\pm 3.5}$ | **90.9**$_{\pm 4.1}$ | **84.4**$_{\pm 2.7}$ | 56.7$_{\pm 0.4}$ | **61.7**$_{\pm 0.8}$ | **2.7** |

Table 29: O.o.d-detection AUC-PR (↑) using different EBMs in a transductive setting.

| | Model | LoC | Ber($\hat{p}$) | $\mathcal{N}(0,1)$ | Homo. |
|---|---|---|---|---|---|
| **APPNP** | EBM | 90.5 | **71.9** | 17.0 | 73.3 |
| | Safe | **91.6** | 55.9 | 33.1 | 73.6 |
| | GEBM | 88.6 | 67.9 | **67.9** | **78.8** |
| **GCN** $\sigma = 0.5$ | EBM | **60.3** | **52.2** | 17.9 | **52.2** |
| | Safe | 58.7 | 50.0 | 31.9 | 50.8 |
| | GEBM | 14.1 | 35.5 | **99.6** | 32.8 |
| **GCN** $\sigma = 1.0$ | EBM | **75.4** | **60.6** | 3.6 | **61.6** |
| | Safe | 74.4 | 53.9 | 20.0 | **61.6** |
| | GEBM | 29.9 | 37.5 | **99.6** | 40.4 |
| **GCN** $\sigma = 2.0$ | EBM | 84.1 | **75.4** | 2.6 | 67.2 |
| | Safe | **89.8** | 60.1 | 17.4 | 69.1 |
| | GEBM | 89.4 | 61.1 | **99.6** | **75.9** |

Table 30: O.o.d. detection AUC for APPNP and GCN with spectral normalization at different weight scales $\sigma$.

