# OpenReview forum: "Energy-based Epistemic Uncertainty for Graph Neural Networks"
_NeurIPS.cc/2024/Conference — NeurIPS 2024 spotlight_

### Official Review · Reviewer_VZ1v · 2024-06-25

**Soundness:** 3
**Presentation:** 3
**Contribution:** 2
**Rating:** 6
**Confidence:** 3

**Summary:**

This paper explores the challenges associated with quantifying uncertainty in Graph Neural Networks (GNNs), particularly in domains involving interconnected data such as graphs. The authors propose a novel method called GEBM, which aggregates energy at various structural levels. This approach enhances predictive robustness and improves performance in distinguishing between in-distribution and out-of-distribution data across different anomaly types and datasets.

**Strengths:**

1. The paper introduces a novel method based on energy considerations. The authors provide sufficient background and related work on the topic, making the main idea accessible even to readers unfamiliar with the subject.
2. Theoretical proofs and theorems are adequately presented to support the proposed methodology. The paper includes comprehensive experimental settings and achieves state-of-the-art performance across various tasks.
3. The paper is well-organized and well-written, enhancing clarity and readability.

**Weaknesses:**

1. The experiments predominantly focus on node-level tasks. It would be beneficial to explore how the proposed method performs on graph-level tasks as well.
2. The paper lacks an ablation study specifically on different types of energy used in the GEBM model. Understanding which energy components are critical remains unclear.

**Questions:**

1. From the equations presented, it appears that this method is not limited to graphs. Are there potential challenges in applying this method to tasks beyond graph-related domains?

**Limitations:**

The limitations are discussed in this paper. GEBM is a post hoc epistemic estimator; it does not improve aleatoric uncertainty or its calibration.

---

> ### Author Rebuttal · Authors · 2024-08-06
>
> We thank the reviewer for their thorough review and are happy that they like our paper overall.
>
> **Weakness 1, Graph-Level Tasks**: Indeed, transferring GEBM to graph-level tasks is an interesting avenue: The energies proposed by our framework are designed with a node-level objective in mind. If suitable knowledge of how graph-level properties relate to individual nodes, GEBM would certainly offer a powerful framework to combine them effectively. Devising an aggregation scheme would be a daunting problem that is not straightforward to solve, in general, and beyond the scope of our work. We will mention that direction in the future work, L.332: “Future work may build upon the framework of aggregating energies at different scales for graph-level tasks by defining and adequately aggregating node-level energy at different scales into a per-graph measure.”
>
> **Weakness 2, Ablation of Energy Types**: There seems to be a misunderstanding regarding this ablation study. We provide ablations regarding the different energy types (local, group, and independent) for all distribution shifts and datasets in Tables 3 and 26 of our paper. We will highlight that this is an ablation of the individual energies more clearly by adding the following to our paper, L297: “Each of the former corresponds to only using one of the energies that the GEBM framework is composed of.” We hope that this sufficiently avoids this potential misunderstanding for future readers.
>
> **Question 1, Non-Graph Domains**: It is true that energy-based models are not limited to the graph domain, in general, and have also successfully been applied to i.i.d. domains [1]. As we showcase in Section 5.3 (in particular Table 3) of our paper, the effectiveness of GEBM primarily comes from aggregating energies at different structural scales in the graph. This is inherently bound to graph problems. However, we agree with the idea of the reviewer that also other domains may benefit from the paradigm of composite energy at different resolutions: One could, for example in the domain of time series, define energies at different frequencies and aggregate them into a single measure similar to GEBM. Also, density-based regularisation is likely to improve EBM-based uncertainty in other domains. The key challenge would be to develop energies that arecapable of capturing anomalies of different types. How these terms would look most likely depends on the problem. We will acknowledge this nice idea in the future work section of our paper, L.332: “The effectiveness of GEBM also motivates the development of aggregate energy-based epistemic uncertainty for other domains. We leave transferring our approach to non-graph problems for future work.”
>
> Overall, we want to thank the reviewer for their review. We hope that we could clarify the misunderstanding regarding the ablation asked in Weakness 2 and appropriately address how GEBM could be extended to non-node-level or even non-graph-level tasks in the future work section of our revised manuscript.
>
> References:
>
> [1]: Liu, Weitang, et al. "Energy-based out-of-distribution detection." *Advances in neural information processing systems*33 (2020): 21464-21475.

---

> > ### Comment · Reviewer_VZ1v · 2024-08-11
> > **Reply to the rebuttal**
> >
> > Thanks for your rebuttal. My concerns have been partially addressed. I would like to maintain the rating.

---

> > > ### Author Response · Authors · 2024-08-12
> > >
> > > We are glad that we could resolve some concerns and want to thank the reviewer again for the time spent on the review, in particular for pointing out interesting directions for future work.

---

### Official Review · Reviewer_Q6TT · 2024-07-08

**Soundness:** 3
**Presentation:** 3
**Contribution:** 3
**Rating:** 7
**Confidence:** 3

**Summary:**

The paper defines an integrable (regularized) energy function to capture epistemic uncertainty via energy a pretrained model. The energy function is a function of the logits so the method is a post-hoc model agnostic. The authors define a diffusion-based hierarchical energy propagation (structure agnostic + local diffusion + group diffusion) which both leads to quantification of uncertainty in graphs, and an evidential model prediction.

**Strengths:**

I count the regularization of the energy function and the theory behind it as a strong point of the paper.

Also defining an energy-based model for graphs to address uncertainty is a strong starting point for uncertainty quantification on graphs which is really under-explored by the current time.

I also see that the authors have provided a complete experimental setup (with some minor exceptions which I addressed in the weaknesses).

In total, I find this paper a strong paper, however I believe that it can be stronger with a better flow of the text, and more contribution in theory for the quality of the UQ.

**Weaknesses:**

1. The authors evaluate their method structurally (line 227) via leaving least homophilic nodes or low-page rank centrality as o.o.d. This seems like there is an implicit assumption of homophily in the graph which is not stated anywhere. In other words, I assumed that the propositions, or setup I should see an assumption of "homophily $\ge$ some constant on expectation...". I see that they referred to this assumption in the limitations, but it is better to be mentioned somewhere in other sections as well.

2. *Clearity:* However successful the method is, I do not see a clear intuition on why these three levels of propagation should be combined and why all are aggregated with weight = 1. For this, I expected an intuitive introductory experiment to clearly show what happens to a node before and after each certain diffusion. More importantly, I see theory to show that the regularization $\pm$ diffusion is integrable, or is not infinity anywhere, but I did not find any theory behind why the approach leads to a good uncertainty quantification in the end; like comparing it with an oracle or finding bounds on the distance from unseen ground truth probability. I also did not see any synthetic experiment in this direction which might help a lot.

3. *Experiments:* (1) I see that in scores like ECE and Brier score the method is not the best. Is there any intuitive explanation for why this is the case? I also strongly recommend the authors to mention that in the limitations of the paper. (2) I see the absence of study on models that have diffusion at the probability space; e.g. APPNP. If the GEBM can improve upon these methods then clearly there is some additional information passed in the energy domain.

4. *Minor Typos:* (1) Line 237 the term "fitted" is used twice. More important (2), in Line 163 there should be "for $\boldsymbol{x} \in {Q}_l$" added somewhere to show that the definition of $f(\boldsymbol{x})$ is limited to that polytope.

**Questions:**

1. Building on weaknesses no. 1. (*W1*), do you assume some homophily property like $\mathbb{P}_{v_i \sim v_j}[y_i = y_j] \ge p$? Is heterophily graph a theoretical limitation of your approach? If yes can you elaborate on the theoretical insight behind it? Note that I see the heterophily is mentioned as a limitation but mostly I can not find a sound explanation of why it is other than just leaving it as an assumption.

2. In evaluation with feature perturbations, why didn't the authors use a random XOR shift instead of a total replacement of features? In that case, you have control over the magnitude shift; intuitively I expect the uncertainty to grow with a correlation to the perturbation probability, but here I can just see the endpoints of the experiment I just mentioned -- fully perturbed node and original node. In general, you can also define the feature shift by randomly selecting from the noise or the original features and controlling the randomness as the magnitude of the shift.

3. In Fig. 2. (robust evidential inference), I can not understand why the result is non-trivial. If the graph is homophily, a simple diffusion over predicted probabilities with a strong coefficient can have a significant denoising effect in the prediction. This is especially in case the perturbation is sparse. What is the model evaluated in Fig. 2? Does it have a similar enhancement in robustness for models that already have a diffusion step like APPNP?

**Limitations:**

I think the limitations are mentioned clearly.

---

> ### Author Rebuttal · Authors · 2024-08-06
>
> We thank the reviewer for their in-depth review and interesting questions and are happy to see that they find our paper strong.
>
> **Weakness 1 & Question 1, Homophily Assumption**: We assume homophily throughout our work. We agree that is benificial to explicitly state this assumption early on and add to our background section, L.66: “Our work is concerned with homophilic graphs, i.e. edges are predominantly present between nodes of the same class.”
>
> All diffusion processes we study (Appendix C.6) rely on homophily. Formally quantifying the effectiveness on these operators for different problems with respect to an explicit homophily value is difficult. There are, however, empirical studies on the performance of GNNs that rely on these diffusion processes [2]. We have addressed this in L.197: “Intuitively, this is a smoothing process that relies on the in-distribution data to be homophilic. In case of non-smooth (heterophilic) data, the energy will be high for in-distribution data which is undesired behavior.”. We believe that formally studying the homophily assumption for graph diffusion is future work.
>
> **Weakness 2, Energy Types and Weights:** Uniform weights do not assume prior knowledge about the problem. We believe that tuning these weights on out-of-distribution data may be unrealistic, as such data is unavailable in real scenarios. The effectiveness of even the most uninformed (uniform) choice shows GEBM’s merits on different distribution shifts without any tuning.
>
> We agree that the paper benefits from a motivating experiment and intuition for each energy type.
>
> **1. Independent Energy** uses no diffusion and therefore targets anomalies at individual nodes. Formally, this is motivated as diffusing energy from non-anomalous low-energy neighbors would decrease the energy of an anomalous node.
>
> **2. Group Energy** uses diffusion to increase the energy of a node if anomalous high-energy neighbors are present and is fit to detect cluster-level anomalies. A similar formal argument can be made here.
>
> **3. Local Energy** - in contrast to Group Energy - does not aggregate class-wise energy before the diffusion. Therefore, it aggregates local average per-class evidence for a node and can identify evidence conflicts arising from structural anomalies.
>
> We devise synthetic experiments that support the intuition behind all three energy types in Figure 4 (global response): First, we induce artificial structural anomalies into real data by iteratively introducing a left-out-class (anomalous) cluster node-by-node. Group Energy shows the highest increase. Second, we induce per-node anomalies into a synthetic SBM graph and increase the magnitude, and observe Independent Energy to be the most sensitive. Lastly, we sample SBMs of varying heterophily (measured as the log ratio of inter-class to intra-class edge probabilities) and confirm that only Local Energy detects the structural anomaly. We propose to add this synthetic experiment along with the intuitive explanation to Section 4 of the updated paper. We would be happy to hear the reviewer’s opinion on that.
>
> **Weakness 3, Calibration and APPNP**: GEBM is a post-hoc estimator and does not affect the calibration of the backbone classifier and we report its ECE and Brier score for completeness only. We will explicitly highlight this more clearly in the limitations section, L.330: “The GCN backbone used in this work does not consistently achieve the strongest performance in both tasks.”. We also want to point out the calibration could easily be improved e.g. with temperature scaling [1]. We view this as outside the scope of our work as it is unrelated to GEBM’s epistemic uncertainty.
>
> We also add APPNP to the ablation of model backbones (Table 1, global response). GEBM is the only method that consistently detects all families of distribution shifts. This confirms that multi-scale uncertainty also improves on models based on just diffusion. We are thankful for that pointer as it makes a strong point in favor of GEBM.
>
> **Weakness 4, Typos:** Thanks, we fixed that typo.
>
> **Question 2, XOR Feature Shift**: Thanks for the nice proposal: We experiment with this XOR shift (Figure 5, global response) and observe that GEBM is the only method that reliably provides good performance with an increasing fraction of perturbed features.
>
> **Question 3, Advantage of Evidential Inference**: The model evaluated in Figure 3 is a GCN. We agree that diffusion in general can denoise anomalous predictions. The key advantage of evidential methods is that with increasing anomaly severity the evidence approaches zero while, for example, logits are provably likely to approach infinity. Therefore, a fixed amount of denoising steps is sufficient to recover from arbitrarily severe corruption while non-evidential methods require more diffusion iterations the stronger the anomaly is. As requested, we can provide evidence for this through an experiment with APPNP (Figure 4, global response). The evidential interpretation of GEBM on an APPNP is significantly more robust than APPNP alone. We add to the paper, L.289: “The advantage of this evidential perspective is that the evidence approaches zero for increasingly severe distribution shifts. Therefore, a fixed number of diffusion steps effectively counteracts the influence of anomalous neighbors when making predictions for a node.”
>
> We are very grateful for the pointers. We believe that the changes to our manuscript and additional experiments help to clarify and consolidate the core assumptions and results of our work.
>
> **References:**
>
> [1]: Chuan Guo, Geoff Pleiss, Yu Sun, Kilian Q. Weinberger. “On Calibration of Modern Neural Networks”. International Conference on Machine Learning (ICML) 2017.
>
> [2]: Palowitch, John, et al. "Graphworld: Fake graphs bring real insights for gnns." *Proceedings of the 28th ACM SIGKDD conference on knowledge discovery and data mining*. 2022.

---

> > ### Comment · Reviewer_Q6TT · 2024-08-14
> >
> > Thanks for the detailed reply.
> >
> > My concerns were partially addressed. With the informative reply from the authors, I find this paper an acceptable and strong study. This is why I increase my score.

---

> > > ### Author Response · Authors · 2024-08-14
> > >
> > > We are glad that we could address the reviewer's concerns and believe that our paper benefits from additional experiments prompted by their feedback. Thank you for the very helpful review!

---

### Official Review · Reviewer_nGhJ · 2024-07-13

**Soundness:** 3
**Presentation:** 3
**Contribution:** 2
**Rating:** 6
**Confidence:** 4

**Summary:**

This paper introduces a method for post-hoc epistemic uncertainty estimation in logit-based Graph Neural Networks (GNNs) by aggregating energy scores at different levels, including node, local, and group levels. Extensive experiments show the effectiveness of the proposed framework.

**Strengths:**

1.	The paper rigorously evaluates the proposed method under various experimental conditions, such as out-of-distribution (OOD) selection, different GNN backbones, and both inductive and transductive evaluation settings.
2.	It comprehensively aggregates uncertainties at multiple levels in the graph, including node-level uncertainties, class-specific neighbor information, and propagated energy through diffusion.
3.	The manuscript is well-structured, with a clear presentation of concepts, logical flow, and detailed preliminary knowledge.

**Weaknesses:**

1.	The paper lacks a detailed discussion on the selection of hyperparameters, especially for the diffusion module $P_A$. Specifics about the parameters $\alpha$ and $t$ mentioned in Appendix C are not sufficiently discussed. Including ablation studies on different graph diffusion architectures, such as label propagation referenced in the Appendix or APPNP used in the GPN paper, would enhance the paper.
2.	The paper states that common GNNs suffer from overconfidence due to their similarity to findings on ReLU neural networks[1]. However, literature [2] [3] suggests that predictions from shallow GNNs are typically under-confident. The paper will benefit from evidence on the over-confidence issue of GNNs.
3.	Section 4.4 discusses the relationship between energy scores from logit-based classifiers and total evidence in evidential models. The paper lacks an explanation for why the proposed model outperforms evidential models in epistemic uncertainty prediction, particularly how it addresses the feature collapsing issue in density-based models [4].

[1] Matthias Hein, Maksym Andriushchenko, and Julian Bitterwolf. Why relu networks yield high-confidence predictions far away from the training data and how to mitigate the problem. In Proceedings of the IEEE/CVF conference on computer vision and pattern recognition, pages 41–50, 2019.

[2] Wang, Xiao, Hongrui Liu, Chuan Shi, and Cheng Yang. "Be confident! towards trustworthy graph neural networks via confidence calibration." Advances in Neural Information Processing Systems 34 (2021): 23768-23779.

[3] Wang, Min, Hao Yang, Jincai Huang, and Qing Cheng. 2024. “Moderate Message Passing Improves Calibration: A Universal Way to Mitigate Confidence Bias in Graph Neural Networks”. Proceedings of the AAAI Conference on Artificial Intelligence 38 (19):21681-89. https://doi.org/10.1609/aaai.v38i19.30167.

[4] Mukhoti, Jishnu, Andreas Kirsch, Joost van Amersfoort, Philip HS Torr, and Yarin Gal. "Deep deterministic uncertainty: A simple baseline." arXiv preprint arXiv:2102.11582 (2021).

**Questions:**

1.	How does the paper perform inductive training on the GCN backbone when OOD nodes and edges are excluded during the training phase? Does it use graph sampling or data augmentation techniques?
2.	In corollary 4.3, what is meant by ' any $x\in \mathbb{R}^d$ ’? Please provide a precise range for $x$ or probability.
3.	In the Equation (9), the regularized energy from three structural scales equally contributes to the final energy score. Table 3 shows varying impacts of energy at these scales. Why was the decision made to use equal weighting?
4.	There are inconsistencies between some model names mentioned in Section 5.1 and those in the tables.
5.	What are the differences in distribution shifts used in this paper compared to those in GPN or GNNSafe, and why did the authors make these changes?

**Limitations:**

YES

---

> ### Author Rebuttal · Authors · 2024-08-06
>
> We thank the reviewer for their very thorough review and specific pointers for improvements. We are happy that the reviewer likes our rigorous evaluation and structure.
>
> **Weakness 1, Ablations on Diffusion**: We ablate $t$ and $\alpha$ and the diffusion operator in Figure 1 (global response) and report performance on leave-out-classes, feature perturbations, and the homophily shift. Label Propagation is used per default in GEBM and GNNSafe while APPNP and GPN use symmetric diffusion. As expected, low $t$ and high $\alpha$ aid node-level anomaly detection while the opposite holds for cluster and local shifts. Label-Propagation achieves satisfactory performance over the entire range of diffusion hyperparameters which justifes using it in our experiments. In particular, we did not tune any hyperparameters for good o.o.d.-detection. As stated in Appendix C.6, it does not bias the energy toward high-degree nodes which explains its advantages over the other diffusion types. It performs well over a broad range of hyperparameters making GEBM less sensitive to those.
>
> **Weakness 2, Evidence on GNN Overconfidence**: Previous work indeed finds that GNNs are under-confident on in-distribution test nodes. Our claim regards confidence under distribution shifts for which we are not aware of similar studies. We supply evidence for the overconfidence of GNNs in Figure 2 (global response): We find that both logit-based and unregularized energy-based confidence increase far from the training data. This confirms our theoretical analysis and justifies the use of distance-based regularizers. We are grateful for this pointer, and adapt our paper to explicitly disambiguate (L.157): “We remark that previous studies found GNNs to be underconfident on in-distribution data […] while the aforementioned issue of overconfidence arises from high distance to training data induced by a distribution shift (see Appendix …).”
>
> **Weakness 3, Advantage over Evidential Methods and Feature Collapse**: This is an interesting point: We do not claim energy to be the strictly superior choice to evidential methods. The ablation in Table 3 shows aggregating uncertainty at different structural scales is the driving factor in GEBM’s effectiveness: This enables it to outperform the evidential GPN. Transferring this paradigm to evidential methods is an interesting direction for future work which we added to L.332: “While GEBM enables robust evidential inference, future work may build upon its paradigm of aggregating different structural scales in the graph for evidential methods.”. One merit of EBMs is that they offer a theoretically sound and well-motivated way to combine uncertainty at different structural scales.
>
> Regarding feature collapse, we are not aware of any evidence for this phenomenon in GNNs. We expect that strong spectral normalization (small $\sigma$) helps detect severe feature perturbations. We confirm this by ablating spectrally normalized GCNs in Table 1 (global response), but do not find improvements consistent over all shifts. It also decreases performance on other distribution shifts. We conjecture that either feature collapse is not as prevalent in GNNs (potentially due to smoothing) or that L2-based measures as in [1] are not suitable for the graph domain. While beyond the scope of our work, we believe this to be a daunting direction for future work and add to L.332: “Further development regarding the density-based regularizer, for example by studying the effects of feature collapse […], may improve the energy that GEBM is based on.”
>
> **Question 1, Data Augmentation:** We do not perform any data augmentation in our work.
>
> **Question 2, Precise Definition for x**: Thank you for the pointer, this was phrased imprecisely: The statement holds for any x almost surely, i.e. the set of x for which it does not hold (the kernel space of the affine function) has measure zero. We adapt our phrasing to “and any $x \in R^d$ almost surely:”.
>
> **Question 3, Energy Weights**: We choose equal weights as we do not assume prior task-dependent knowledge about which structural scale is more important than others. We see tuning these weights on o.o.d. data as unrealistic since such data is unavailable in real scenarios. We believe that the effectiveness of even the most uninformed (uniform) choice underlines the merits of the paradigm behind GEBM. Practitioners can also adapt these weights to incorporate prior knowledge. If the reviewer thinks changing Equation (9) to account for weights is beneficial, we are happy to adapt the paper accordingly.
>
> **Question 4:** Thank you for pointing it out, we fix this by changing “GNNSafe → GCNSafe” and “Heat → GCN-HEAT”.
>
> **Question 5**: Our suite of distribution shifts is a superset of the GPN benchmark, with the most notable addition being a structural shift. The single-graph distribution shifts of GNNSafe fall into the same categories as our benchmark: First, they, too, use left-out-classes. Second, they interpolate between node features similar to our feature perturbations. While there is no guarantee this provides semantically meaningful features (especially for bag-of-word features), we use different noise distributions to control similarity to in-distribution data by matching its modality. Lastly, GNNSafe uses a fixed SBM to perturb the structure: Our homophily-based shift induces a more realistic structural shift as the o.o.d. nodes are drawn from real data. Our benchmark studies the same types of shifts as both related works pointed out by the reviewer. Should they believe that we miss a relevant setting, we are happy to include it in the updated paper as well.
>
> We again want to thank the reviewer for the interesting and helpful discussion points that helped us to improve the manuscript.
>
> **References:**
>
> [1]: Mukhoti, Jishnu, Andreas Kirsch, Joost van Amersfoort, Philip HS Torr, and Yarin Gal. "Deep deterministic uncertainty: A simple baseline." arXiv preprint arXiv:2102.11582 (2021).

---

> > ### Comment · Reviewer_nGhJ · 2024-08-12
> >
> > Thank you to the authors for their efforts in providing additional experiments and clarifications. They have addressed my concerns, and I have increased my score.
> >
> > Additionally, I agree with most of the points raised in the rebuttal, particularly regarding feature collapsing and under/over-confidence scenarios. I am also interested in exploring the differences and commonalities between energy-based models and evidential-based models.

---

> > > ### Author Response · Authors · 2024-08-14
> > >
> > > We are happy that the reviewer finds the additional ablations and clarifications helpful and we, too, believe it makes the paper stronger. Thank you for the very useful input!

---

### Author Rebuttal · Authors · 2024-08-06

We thank all the reviewers for the time spent providing valuable feedback. We added all experiments and ablations provided in the pdf to the final version of our manuscript. We briefly want to summarise the most relevant additions to the paper. In the individual rebuttals, we propose word-by-word changes for these points.

- We add APPNP as another GNN backbone to our ablation study to show that the success of GEBM is rooted in describing uncertainty at different structural scales rather than simply diffusing the corresponding scores. On APPNP, GEBM outperforms existing methods and enables more robust evidential inference.
- We ablate different diffusion modules and hyperparameters to justify the choice we made for GEBM and find that GEBM is highly effective over a broad range of settings.
- We justify each energy type and provide an introductory synthetic experiment to ease an intuitive interpretation.
- We supplement additional evidence on the over-confidence of GNNs, distinguish it clearly from under-confidence on-distribution data, and elaborate on the role and empirical relevance of feature collapse for these models.
- We provide clear directions for future work that extends to graph-level and non-graph-level domains.

Overall, we are very happy with the constructive input and believe that based on the reviews we were able to further solidify our approach from an intuitive, theoretical, and empirical perspective.

---

### Decision · Program_Chairs · 2024-09-25

**Decision:**

Accept (spotlight)

**Comment:**

The paper proposes a simple and effective post hoc method applicable to any pre-trained GNN that is sensitive to various distribution shifts. The novelty of the proposed method has been recognized by all the reviewers. The paper also provides strong results on datasets with various distribution shifts. The results show the proposed method could improve the generalization performance of existing backbones.

All the concerns from reviewers have been well-solved from the rebuttal. All reviewers support to accept the paper.